# Crystallography, Group Cohomology, and Lieb–Schultz–Mattis Constraints

Chunxiao Liu[1] and Weicheng Ye[2]

[1]*Department of Physics, University of California, Berkeley, California, USA 94720*[*]
[2]*Department of Physics and Astronomy, and Stewart Blusson Quantum Matter Institute,*
*University of British Columbia, Vancouver, BC, Canada V6T 1Z1*[†]

We compute the mod-2 cohomology ring for three-dimensional (3D) space groups and establish a connection between them and the lattice structure of crystals with space group symmetry. This connection allows us to obtain a complete set of Lieb–Schultz–Mattis constraints, specifying the conditions under which a unique, symmetric, gapped ground state cannot exist in 3D lattice magnets. We associate each of these constraints with an element in the third mod-2 cohomology of the space group, when the internal symmetry acts on-site and its projective representations are classified by powers of $\mathbb{Z}_2$. We demonstrate the relevance of our results to the study of U(1) quantum spin liquids on the 3D pyrochlore lattice. We determine, through anomaly matching, the symmetry fractionalization patterns of both electric and magnetic charges, extending previous results from projective symmetry group classifications.

## Contents

[*] chunxiaoliu@berkeley.edu
[†] victoryeofphysics@gmail.com

# I. Introduction

The $k$-dimensional crystallographic groups constitute an important family of infinite groups within group theory. In both mathematical and crystallographic literature, the two-dimensional and three-dimensional crystallographic groups are commonly referred to as wallpaper groups and space groups, respectively. These groups have played a fundamental role in understanding crystal properties and classifying phases of matter. Beyond condensed matter physics, the concept of $k$-dimensional crystallographic groups also appears in various areas of theoretical physics [1–3] and mathematics [4–6].

While the classification of wallpaper and space groups was completed over a century ago [1], and their representation-theoretic properties have been extensively documented in crystallographic references, the calculation of group cohomology and other homological algebraic properties for these crystallographic groups remains largely incomplete. The main challenge in deriving these properties arises from the infinite order and complex group structures of crystallographic groups, which make it difficult to implement the methods designed for finite group calculations in an efficient way [8–10].

From a physical perspective, the homological algebraic properties of crystallographic groups can greatly help understand the symmetry properties of phases of matter in crystals. Of particular importance is the concept of *quantum anomalies* [11]. Roughly speaking, quantum anomaly refers to the obstruction to having a unique, symmetric, gapped ground state, and different anomalies are captured by distinct elements of the cohomology of the symmetry group [12]. For symmetries acting on a lattice system, a prototypical quantum anomaly is called Lieb–Schultz–Mattis (LSM) anomaly [13, 14], which exists in a lattice system with on-site internal symmetry such that the microscopic, on-site degrees of freedom carry projective representations of the internal symmetry. For example, in 1D, LSM anomaly is present in a spin-1/2 chain with translation and SO(3) rotation symmetry.

Quantum anomaly is important because of the principle of *anomaly matching*: in a lattice system, consider an infrared (IR) theory that emerges on the lattice with some emergent low energy degrees of freedom, the quantum anomaly of this IR theory must match the anomaly present in the original, microscopic (UV) theory. As such, one can ask how the UV symmetries (including crystalline symmetry $G$ and internal symmetry) act on the emergent IR degrees of freedom, and this action is heavily constrained by anomaly matching [15, 16]. Such data of symmetry actions (referred to as "quantum numbers" in physics) can then be numerically tested, providing additional information regarding phases on the lattice [17–19]. This makes anomaly matching a quite powerful theoretical tool besides conventional methods of detecting phases of matter.

To implement the program of anomaly matching, we need to acquire three pieces of data: (1) the UV anomaly of the lattice system; (2) the IR anomaly; and (3) the check that the UV and IR anomalies match. In the context of a spin-1/2 lattice magnet, where the UV anomaly is of the LSM type, the anomaly information is encoded in the mod-2 cohomology of crystallographic groups. Additionally, knowing the complete ring structure of mod-2 cohomology is essential in performing anomaly matching.

———

[1] For an interesting classification of 3D space groups using orbifolds, see [7].

A related matter is the classification of symmetry-protected topological phases (SPTs) protected by crystalline symmetries (crystalline SPTs), where the homological algebraic aspects of crystallographic groups also play a crucial role. Bosonic crystalline SPTs are classified by the group cohomology of crystalline groups [20, 21]. On the fermionic side, while fermionic crystalline SPTs are classified by bordism groups [22–25], the mod-2 cohomology ring (as a module of the Steenrod algebra) still serves as input to the Adams spectral sequence calculation. Therefore, from the perspectives of both quantum anomalies and crystalline SPTs, a deep understanding of the mod-2 cohomology is of paramount importance.

In this work, we conduct a comprehensive study of the homological algebraic properties of all 3D space groups. Taking advantage of recent progress in computational homotopy [9], we systematically obtain the mod-2 cohomology ring for these groups. Furthermore, we explicitly give the standard inhomogeneous cochain expressions for those cocycles of degree 3 or lower. As the mathematical foundation of the computation, we explore the entries and maps in the Lyndon–Hochschild–Serre (LHS) spectral sequence, and prove that the $\mathbb{F}$ cohomology ring of any crystallographic group $G$ must be finitely generated with a finite number of generators and relations. We also estimate the highest degree at which a generator or a relation can appear for 3D space groups. These determine the maximal degree that one needs to investigate in order to generate the full expression of the mod-2 cohomology ring.

Next, we turn to an important physical application of these mod-2 cohomology results: the LSM constraints and LSM anomaly. We connect LSM constraints in 3D with crystal lattice structures, and explicitly identify the cohomology element for each of these LSM constraints for all the 230 space groups using the mod-2 cohomology rings that we computed. Finally, we show an application of these data by understanding the symmetry actions on the emergent IR degrees of freedom for U(1) quantum spin liquids on the 3D pyrochlore lattice, and compare the results with projective symmetry group (PSG) calculations.

## A. Statement of main results

We now present a summary of the main results and outline the organization of the paper.

1. We prove that the $\mathbb{F}$ cohomology ring of any crystallographic group $G$ must be finitely generated with a finite number of generators and relations.

2. We obtain the mod-2 cohomology ring for all (but one) 230 3D space groups using the methods summarized in Sec. IV A. These cohomology rings are collected in Appendix F for all 230 groups. We give an explicit expression for each of the 1-, 2-, and 3-cocycle functions of the mod-2 cohomology of space groups (except for the 3-cocycles of groups No. 225, 227, and 229). These are collected in our online Github files [26].

3. To achieve a complete characterization of LSM anomaly, to each Irreducible Wyckoff Position (IWP, defined in Sec. II B) of every 3D space group we associate a unique element in the third cohomology of the space group with $\mathbb{Z}_2$ coefficient. These data are listed in the IWP Table for each of the 230 groups, collected in Appendix F. A cohomological-operational characterization of all these elements is given in Statement 18.

4. The data of IWP allow us to assemble the complete statement of the LSM constraints for 3D lattice magnets when the on-site degrees of freedom carry a projective representation of the internal symmetry group, and the projective representations are classified by powers of $\mathbb{Z}_2$. The statement is given in Statement 17.

5. In Sec. VI, we demonstrate the physical significance of our result by performing anomaly matching for $U(1)$ quantum spin liquids on the pyrochlore lattice. We obtain the symmetry fractionalization of electric and magnetic charges in Eq. (86) and Eq. (85), respectively, for a type of symmetry actions. The result of the symmetry fractionalization of fermionic electric charges is consistent with the PSG calculation in Ref. [27].

The paper proceeds as follows. In Sec. II, we give a brief introduction of crystallographic groups and the concept of IWP, and set up the notation that will be used for the rest of the paper. In Sec. III, we state and prove several theorems for cohomology of crystallographic groups, which serves as the mathematical foundation of our codes. In Sec. IV, we outline our methods to obtain the mod-2 cohomology of 3D space groups, in particular the ring structure, and discuss some features of our results. In Sec. V, we review existing LSM constraints in 2D and its connection to cohomology. Then we move on to 3D, state the various LSM constraints from IWPs and spell out the algorithm we use to associate each IWP of space groups $G$ with an element in $H^3(G, \mathbb{Z}_2)$, with several notable examples included in Sec. V E. In Sec. VI, we apply LSM anomaly and anomaly matching to the understanding of $U(1)$ quantum spin liquids on the pyrochlore lattice and compare our results with existing PSG calculations. We close the paper with open questions and outlooks. Several appendices are included to provide the necessary information for 3D space groups and its cohomology, with the complete enumeration of results for all 230 space groups given in Appendix F.

Many of the results are obtained with the help of the softwares GAP [8], SageMath [28], and Mathematica [29]. The codes are available on Github [26].

## II. Basics of crystallographic groups

In this section, we introduce the basic concepts of crystallographic groups, including the notion of Irreducible Wyckoff Positions (IWPs), which will be central to the description of Lieb–Schultz–Mattis constraints. We also set up the notation that will be used throughout this paper.

### A. Crystallographic groups in 3D

Crystallographic groups are groups consisting of symmetry operations of crystals that leave the crystal structure invariant [30]. We first give its formal definition:

**Definition 1.** *(Crystallographic groups of dimension k. [9]) Let $G$ be a group of invertible affine transformations of $\mathbb{R}^k$, such that any element $g \in G$ has the form $g\colon \mathbb{R}^k \to \mathbb{R}^k$, $\boldsymbol{x} \mapsto A\boldsymbol{x} + \boldsymbol{b}$, where the matrix $A \in GL_k(\mathbb{R})$ and the vector $\boldsymbol{b} \in \mathbb{R}^k$. g is called pure translation if $A$ is the identity matrix. Define $T$ as the normal subgroup consisting of pure translations in $G$. The group $G$ is a crystallographic group of dimension $k$ if $T$ is a free abelian group of rank $k$ and the quotient group $P := G/T$ is finite.*

The translation group $T$ is, by definition, isomorphic to $\mathbb{Z}^k$. The quotient group $P$ is called *point group* associated with $G$. The three groups fit into the short exact sequence

$$1 \to T \to G \to P \to 1. \tag{1}$$

The short exact sequence Eq. (1) offers a complementary view for the three groups $T$, $G$, and $P$. Given $P$ as a finite group, $T$ is an integral representation of $P$ whose action is defined by conjugation

$$\rho\colon P \times T \to T, \quad (p, t) \mapsto s(p) \cdot t \cdot s(p)^{-1}, \tag{2}$$

with $s(p)$ any pre-image of $p$ in $G$. $G$ is then thought of as the group extension of $P$ by $T$, characterized by the integral representation $\rho$ and an element $\omega$ of the second cohomology group $H^2_\rho(P, T)$.

Unless otherwise stated, we will always use $G$ to denote a crystallographic group, $P$ to denote its associated point group, and $T$ to denote the group of pure translations $\mathbb{Z}^k$. Sometimes we will also use $\mathcal{G}$ to denote the full symmetry group of a physical system, including lattice symmetries and internal symmetries.

Going to 3D, the classification of space groups was done separately by Fedorov and Schönflies in 1891. Treating $P$ as abstract groups, we know there are 18 possibilities in 3D

$$\begin{aligned}
&\mathbb{Z}_1, \mathbb{Z}_2, \mathbb{Z}_2^2, \mathbb{Z}_2^3, \mathbb{Z}_4, \mathbb{Z}_4 \times \mathbb{Z}_2, Dih_4, Dih_4 \times \mathbb{Z}_2, \\
&\mathbb{Z}_3, \mathbb{Z}_3 \times \mathbb{Z}_2, \mathbb{Z}_3 \times \mathbb{Z}_2^2, Dih_3, Dih_3 \times \mathbb{Z}_2, Dih_3 \times \mathbb{Z}_2^2, \\
&A_4, A_4 \times \mathbb{Z}_2, S_4, S_4 \times \mathbb{Z}_2,
\end{aligned} \tag{3}$$

where $Dih_n, A_4, S_4$ stands for dihedral group with $2n$ elements, the alternating group on 4 letters, and the symmetric group on 4 letters.

The abstract classification of point groups do not capture all the features of crystal symmetry. Depending on the symmetry operations (rotation, mirror reflection, inversion, etc.) on $\mathbb{R}^3$, the abstract point groups are further distinguished into 32 point groups. The detailed information about the 18 abstract point groups and the 32 point groups are listed in Table I.

To name the generators of the point groups, we use the following convention:

- $C_n$: $n$-fold rotation $C_n$

- $M$: mirror reflection

- $I$: inversion

- $G$: glide

- $S_n$: $n$-fold screw

- $\overline{C}_n$: $n$-fold rotoinversion (only $\overline{C}_4$ will explicitly appear).

The 32 point groups, combined with the 14 Bravais lattice types, determine a total of 73 *arithmetic crystal classes*, in one-to-one correspondence with the distinct integral representations $\rho$ (see Eq. (2)). For each arithmetic class, the 2nd cohomology $H^2_\rho(P, T)$ classifies the distinct group extensions. The arithmetic classes together with $H^2_\rho(P, T)$ determines a total number of 219 (non-isomorphic) abstract space groups as possible extensions of $P$ by $T$. The detailed information about the arithmetic classes and the extension classes is given in Table III. Each arithmetic class contains a split extension,

for which the space group is a semidirect product $G = T \rtimes P$. In crystallography literature, the split crystallographic groups are called *symmorphic* groups. Therefore, there are in total 73 symmorphic groups, one for each arithmetic crystal classes. These symmorphic groups are labeled with a "$\rtimes$" in the third column of Table IV.

Finally, out of the 219 abstract space groups, 11 of them describe crystal structures with a designated chirality [2]. The following pairs of isomorphic space groups are introduced to further distinguish the two chiralities

$$
\begin{aligned}
& 76\ (P4_1) \cong 78\ (P4_3), && 91\ (P4_122) \cong 95\ (P4_322), && 92\ (P4_12_12) \cong 96\ (P4_32_12), \\
& 144\ (P3_1) \cong 145\ (P3_2), && 151\ (P3_112) \cong 153\ (P3_212), && 152\ (P3_121) \cong 154\ (P3_221), \\
& 169\ (P6_1) \cong 170\ (P6_5), && 171\ (P6_2) \cong 172\ (P6_4), && 178\ (P6_122) \cong 179\ (P6_522), \\
& 180\ (P6_222) \cong 181\ (P6_422), && 212\ (P4_332) \cong 213\ (P4_132),
\end{aligned}
\tag{4}
$$

and this brings the final number of space groups to 230.

In Appendix F, for each group, we list the generators and their actions on the coordinate systems. Our choice of the coordinate systems is the same as the "Standard/Default Setting" on Bilbao Crystallographic Server [31] (these "Standard/Default Settings" are always one of the coordinate setups given in the International Tables for Crystallography (ITC) [32]). The numbering of the 230 space groups follows the standard crystallography numbering [32].

## B. (Irreducible) Wyckoff position and lattice homotopy

In this subsection, we introduce the concept of Wyckoff positions and irreducible Wyckoff positions (IWPs).

Even within a given crystallographic symmetry setting, arranging degrees of freedom (spins) in different configurations can yield distinct lattices (which share the same crystalline symmetry). A crystallographic lattice $\Lambda$ is a set of discrete points in $\mathbb{R}^k$ invariant under the crystallographic group $G$. We will assume that there are some physical degrees of freedom (e.g. spins) located at each site of the lattice, and they are acted upon by some internal symmetry [3]. To fully characterize all possible lattice structures resulting from these arrangements, we introduce the concept of the *Wyckoff position*. Suppose $\Lambda$ contains a point $s \in \mathbb{R}^k$, we need to include all points $g.s$ that are related to $s$ by a crystalline symmetry $g \in G$. Still, there are some elements $g \in G$ such that they keep $s$ invariant, and they form a subgroup of $G$ that must be one of the point groups. We call this subgroup the *little group* of the site $s$ (which is also called the *stabilizer* of the site $s$ in the math literature).

**Definition 2.** *(Wyckoff position [33]) A Wyckoff position of a crystallographic group $G$ is an equivalence class of points in Euclidean space, such that their associated little groups are conjugate subgroups of $G$.*

A Wyckoff position can be understood as a set of points (or orbits) on a crystal lattice that are transformed in the same way under the crystallographic group. For a given crystallographic group, placing identical physical degrees of freedom on different Wyckoff positions—or even on multiple Wyckoff positions simultaneously—will result in distinct lattice structures. The little group of a Wyckoff position does not fully specify the Wyckoff position: there can be distinct Wyckoff positions with isomorphic little groups. Still, the little group of a Wyckoff position is its most important property.

In addition, to analyze symmetry actions on the lattice structure and extract anomaly from these symmetry actions, we are free to symmetrically move degrees of freedom around in a continuous way. This operation is part of so-called *lattice homotopy* [34, 35]. This gives rise to the concept of *irreducible* Wyckoff positions.

**Definition 3.** *(Irreducible Wyckoff Position, IWP) Given a Wyckoff position, consider the closure of all points belonging to the Wyckoff position. If the little group of the Wyckoff position is not a proper subgroup of any other points in the closure, we will call the Wyckoff position irreducible.*

Colloquially, a Wyckoff position is reducible if we can *symmetrically* tune the points in the Wyckoff position to a nearby "high symmetry points", and thereby enhance the little group to a bigger group. To analyze the anomaly of symmetry actions, we are free to symmetrically move all degrees of freedom to these IWPs, and hence simplify the analysis significantly.

For instance, in 2D, consider the wallpaper group $p6m$, generated by two translations $T_{1,2}$, a six-fold rotation $C_6$ and a mirror reflection $M$. This group has three distinct IWPs in total, corresponding to the center of six-fold rotation $C_6$ (a), the center of three-fold rotation $T_1 C_6^2$ (b) and the center of two-fold rotation $T_1 C_6^3$ (c), respectively, as illustrated in Figure 1. When spins are placed on these three IWPs, they form triangular, honeycomb, and kagome lattices. While all three lattices share the same crystalline symmetry $p6m$, the little group at each site differentiates them.

As a 3D example, consider the space group No. 227 ($Fd\bar{3}m$), generated by three translations $T_{1,2,3}$, two-fold rotations $C_2$ and $C_2'$, a three-fold rotation $C_3$, a mirror $M$, and an inversion $I$. This group has four distinct IWPs in total,

---

[2] A left-handed space group is related to the right-handed one by the conjugation of certain affine transformation $\boldsymbol{x} \mapsto A\boldsymbol{x} + \boldsymbol{b}$, such that $A \in GL_n(\mathbb{R})$ has determinant $-1$.

[3] For most of our discussion, we assume that crystalline symmetry operations only permute the physical degrees of freedom without extra effects, and that the internal symmetry is SO(3) whose projective representation is classified by $H^2(\mathrm{SO}(3), \mathrm{U}(1)) \cong \mathbb{Z}_2$. We will address possible generalizations as they become relevant.

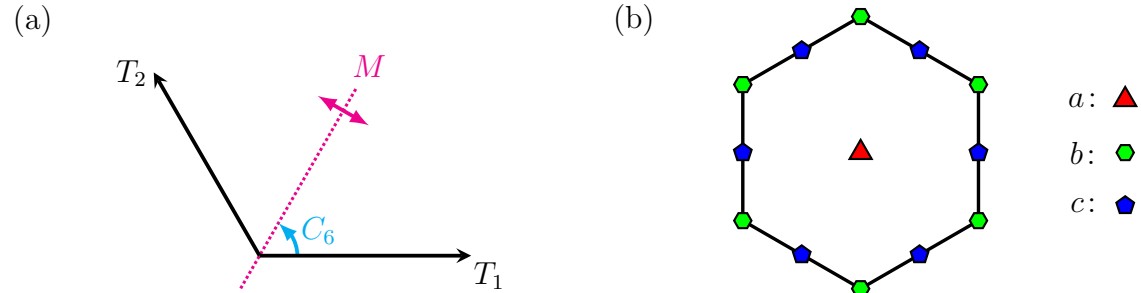

FIG. 1. Panel (a) shows the generators of the wallpaper group $p6m$. In panel (b), the hexagon is a translation unit cell of the wallpaper group $p6m$. It has three IWPs, conventionally labeled by $a$, $b$ and $c$ in crystallography, and they form the sites of the triangular, honeycomb and kagome lattices, respectively.

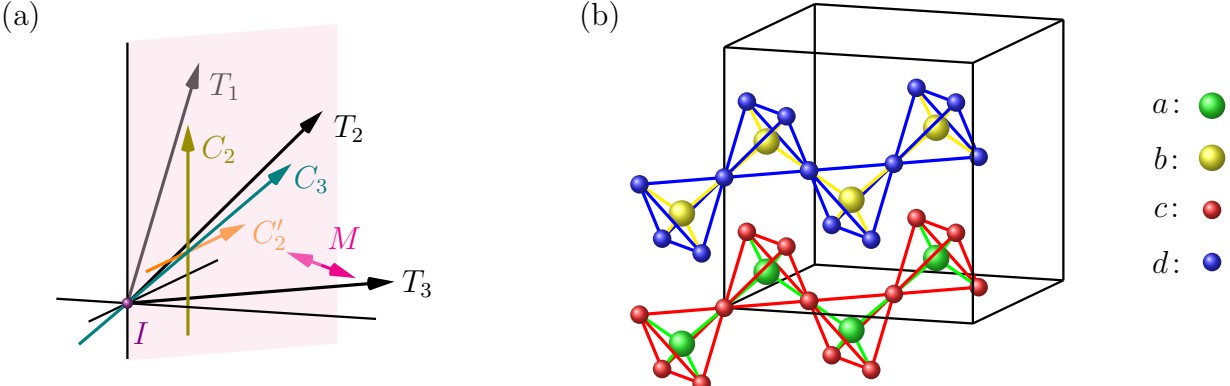

FIG. 2. Panel (a) shows the generators of the space group No. 227 ($Fd\bar{3}m$). In panel (b), the cube is a conventional unit cell of the $Fd\bar{3}m$. It has four IWPs, conventionally labeled by $a$, $b$, $c$, and $d$ in crystallography, where $a$ and $b$ form two sets of diamond lattices and $c$ and $d$ form two sets of pyrochlore lattices.

corresponding to centers of inversions $I$ (c) and $T_1T_2I$ (d), and the intersection points of two pairs of orthogonal two-fold rotations $(C_2, C_2')$ (a), and $(T_3C_2, T_2C_2')$ (b). When spins are placed on one of the inversion centers one gets a pyrochlore lattice, and when spins are placed on one of the intersection points of a pair of orthogonal two-fold rotations one gets a diamond lattice. The symmetry operations and the four IWPs are illustrated in Figure 2.

Besides moving degrees of freedom around, for the analysis of anomaly, we are also allowed to fuse these degrees of freedom together in a symmetric way. This includes the following two possibilities [35, 36]: (1) remove sites that carry linear representation under all symmetry actions (especially the internal symmetry) and (2) combine two sites at the same location into a single site which carries the tensor product representation of the two original sites. For example, when we have on-site internal SO(3) symmetry, we can fuse two spin-1/2 local moments and discard them, since the outcome is a linear representation of SO(3) that does not contribute to the analysis of anomaly.

*Lattice homotopy* is the operation of moving and fusing degrees of freedom according to the rules above. For a given lattice symmetry $G$, (local) physical degrees of freedom in $\mathbb{R}^k$ form equivalence classes under the operation of lattice homotopy, which we call *lattice homotopy classes* [15, 34, 36] and carry the structure of abelian group under local fusion. When the internal symmetries only act on-site and do not mix with crystalline symmetries, the lattice homotopy classes should carry the structure of $PR^n$, where $n$ is the number of IWPs and $PR = H^2(G_{\text{int}}, \text{U}(1))$ labels the projective representation of internal symmetry $G_{\text{int}}$ at each site.

## III. Structure theorems for cohomology of crystallographic groups

In this section, we state and prove several structure theorems for cohomology of crystallographic groups. In Sec. III A, we analyze the LHS spectral sequence for mod-2 cohomology of crystallographic groups and illustrate where a generator and relation can appear from the point of view of LHS spectral sequence through specific examples. Building on this, in Sec. III B, we prove that the $\mathbb{F}$-cohomology ring of any crystallographic groups must be finitely generated as an $\mathbb{F}$-algebra for some field $\mathbb{F}$. This proof is adapted from the proofs for finite groups in e.g. Refs. [9, 37, 38]. The two subsections serve as the mathematical foundation for our codes to calculate the mod-2 cohomology rings.

To collect additional miscellaneous results of cohomology of crystallographic groups, in Sec. III C, we connect the results

of mod-2 cohomology with integral cohomology. Lastly, in Sec. III D, we discuss when the resolution of crystallographic groups admits a periodic resolution.

## A.  Spectral sequence

Although most calculations are carried out without using Lyndon–Hochschild–Serre (LHS) spectral sequence, the LHS spectral sequence remains an invaluable tool in our analysis, since it gives an upper bound on the degree at which new generators and relations can appear. In this subsection, we summarize key facts about the LHS spectral sequence, emphasizing its application to the cohomology of crystallographic groups. For further reading, there are several standard textbooks on spectral sequences, such as Refs. [37, 39, 40]. Many of the formal discussions can be applied to any field $\mathbb{F}$ where $G$ acts trivially, hence we will keep $\mathbb{F}$ and specialize to $\mathbb{Z}_2$ when discussing specific examples.

Given an arbitrary (discrete) group $G$, with a normal subgroup $N \lhd G$ and the quotient group $P = G/N$, and a field $\mathbb{F}$ which $G$ acts trivially, we have a short exact sequence of groups,

$$1 \to N \to G \to P \to 1. \tag{5}$$

Then we have the LHS spectral sequence, which can be written as

$$E_2^{p,q} = H^p(P, H^q(N, \mathbb{F})) \Rightarrow H^{p+q}(G, \mathbb{F}). \tag{6}$$

An important property is the structure of bilinear product in LHS spectral sequence, which we summarize below.

**Theorem 1.** *(Bilinear product of LHS spectral sequence [9, 40]) The LHS spectral sequence Eq. (6) admits bilinear products*

$$E_r^{p,q} \times E_r^{s,t} \to E_r^{p+s,q+t} \tag{7}$$

*for $r \geq 1$ which satisfy the following properties:*

- *Each differential $d_r \colon E_r \to E_r$ satisfies the Leibniz rule*

$$d_r(xy) = (d_r x)\, y + (-1)^{p+q} x\, (d_r y)\,. \tag{8}$$

- *The Leibniz rule ensures that the product on $E_r$ induces a product on $\ker d_r / \operatorname{im} d_r$, which is the same (ring isomorphism) as the bilinear product on $\ker d_r / \operatorname{im} d_r \cong E_{r+1}$.*

- *The bilinear product on the $E_2$ page $E_2^{p,q} \times E_2^{s,t} \to E_2^{p+s,q+t}$ is the composition*

$$H^p\left(Q, H^q(N, \mathbb{F})\right) \times H^s\left(Q, H^t(N, \mathbb{F})\right) \xrightarrow{\cup_1} H^{p+s}\left(Q, H^q(N, \mathbb{F}) \otimes_{\mathbb{F}} H^t(N, \mathbb{F})\right)$$
$$\xrightarrow{\cup_2} H^{p+s}\left(Q, H^{q+t}(N, \mathbb{F})\right) \xrightarrow{\times (-1)^{qs}} H^{p+s}\left(Q, H^{q+t}(N, \mathbb{F})\right), \tag{9}$$

*where $\cup_2$ is induced by the cup product $H^q(N, \mathbb{F}) \times H^t(N, \mathbb{F}) \xrightarrow{\cup} H^{q+t}(N, \mathbb{F})$, and $\cup_1$ is the cup product induced by a diagonal map $\Delta_* \colon R_*^Q \to R_*^Q \otimes R_*^Q$ on a free $\mathbb{Z}Q$-resolution $R_*^Q$.*

- *The bilinear product on the $E_\infty$ page $E_\infty^{p,q} \times E_\infty^{s,t} \to E_\infty^{p+s,q+t}$ is the same as the one induced from the cup product $H^q(G, \mathbb{F}) \times H^t(G, \mathbb{F}) \xrightarrow{\cup} H^{q+t}(G, \mathbb{F})$ on the filtration corresponding to the $E_\infty$ page.*

This makes $E_r^{*,*}$ a double-graded ring on top of the structure of $\mathbb{F}$-algebra. To connect to the cohomology ring $H^*(G, \mathbb{F})$ of $G$, recall that $H^*(G, \mathbb{F})$ can be written as

$$\mathbb{F}[x, y, \dots]/\text{Relations}, \tag{10}$$

where $x, y, \dots$ are called *generators*, and "Relations" generate an ideal in the ring that are quotiented out and treated as zero in the ring. This is called a *free presentation* of $H^*(G, \mathbb{F})$ treated as an $\mathbb{F}$-algebra. For $E_r^{*,*}$, we also have such a free presentation, except that now $x, y, \dots$ are also double-graded. We have the following theorem for the generators and relations of the double-graded ring $E_\infty^{*,*}$ and those for the cohomology ring $H^*(G, \mathbb{F})$ itself.

**Theorem 2.** *If the $E_\infty$ page $E_\infty^{*,*}$ of the LHS spectral sequence admits a free presentation with generators of degree $\leq d$ and relations of degree $\leq n$, then the cohomology ring $H^*(G, \mathbb{F})$ also admits a free presentation with generators of degree $\leq d$ and relations of degree $\leq n$.*

Hence, even though we cannot obtain the full ring structure of $H^*(G, \mathbb{F})$ from the LHS spectral sequence, we can still obtain a lot of information about generators and relations. Moreover, when computing the cohomology ring of $G$, there is an algorithm using LHS spectral sequences to see whether we obtain a complete set of generators and relations [9]. Here we will also use the LHS spectral sequence to deduce similar results for crystallographic groups.

We call the *standard* LHS spectral sequence for crystallographic groups the one associated with the short exact sequence (1), and we have

$$E_2^{p,q} = H^p(P, H^q(T, \mathbb{F})) \Rightarrow H^{p+q}(G, \mathbb{F}). \tag{11}$$

In $k$-dimensions, the translation group $T \cong \mathbb{Z}^k$ has cohomology

$$H^q(T, \mathbb{Z}_2) = \begin{cases} \mathbb{F}^{\binom{k}{q}}, & 0 \le q \le k, \\ 0, & \text{other } q. \end{cases} \tag{12}$$

Hence, for the standard LHS spectral sequence, starting from the $E_2$ page, only the first $(k+1)$-rows have nonzero entries. Since this is a finite number, we have an easy corollary.

**Corollary 3.** *For $k$-dimensional crystallographic groups, the standard LHS spectral sequence collapses at the $E_{k+2}$ page, i.e. $E_{k+2} \cong E_\infty$.*

Specializing to $\mathbb{F} = \mathbb{Z}_2$, the information for the $E_2$ page of the standard LHS spectral sequence for all 230 space groups is listed in the last three columns of Table III, written in terms of the $\mathbb{Z}_2$ ranks for each entry.

For our analysis of the cohomology ring, we do not attempt to obtain the full bilinear product structure for the standard LHS spectral sequences, since usually it suffices to write down each row as a module of the ring $H^*(P, \mathbb{F})$. The top and bottom row is always isomorphic to the ring $H^*(P, \mathbb{F})$ itself, while the middle rows can be a nontrivial module. Especially, we focus on the degree where a nontrivial generator (of the module) appears.

To make our discussion more concrete, we provide two examples, one for the 2D wallpaper group $p4g$ and the other for the 3D space group $Fmm2$. Although these results will not be explicitly used in our automated calculations in GAP, these examples nicely illustrate how generators and relations appear from a spectral sequence point of view.

### 1. Example: p4g

For 2D wallpaper groups, consider $p4g$, which has the most complicated form of mod-2 cohomology ring among all wallpaper groups. The point group of $p4g$ is $D_4$, which as an abstract group is the dihedral group $Dih_4$. $p4g$ is generated by two translations $T_{1,2}$ along two perpendicular directions, a four-fold rotation $C_4$, and a glide reflection $G$ whose reflection axis passes through the rotation center of $C_4$ and bisects the two translation vectors. Acting on the 2D Euclidean space, we have

$$T_1 \colon (x, y) \to (x + 1, y), \tag{13a}$$
$$T_2 \colon (x, y) \to (x, y + 1), \tag{13b}$$
$$C_4 \colon (x, y) \to (-y, x), \tag{13c}$$
$$G \colon (x, y) \to (y + 1/2, x + 1/2). \tag{13d}$$

To write down the standard LHS spectral sequence for $p4g$, first we write down the mod-2 cohomology ring for $D_4$. $D_4 \cong Dih_4$ is generated by a four-fold rotation $C_4$ and a reflection $M$ such that

$$C_4^4 = 1, \quad M^2 = 1, \quad MC_4M = C_4^{-1}, \tag{14}$$

and every element in $D_4$ can be written as $C_4^c M^m$ with $c \in \{0, 1, 2, 3\}, m \in \{0, 1\}$. The mod-2 cohomology ring of $D_4$ is

$$H^*(D_4, \mathbb{Z}_2) = \mathbb{Z}_2[A_c, A_m, B_\alpha]/(A_c^2 + A_c A_m), \tag{15}$$

where the generators $A_c$, $A_m$ and $B_\alpha$ have the following explicit cochain representative in the bar resolution,

$$A_c(C_4^c M^m) = c, \quad A_m(C_4^c M^m) = m, \quad B_\alpha(C_4^{c_1} M^{m_1}, C_4^{c_2} M^{m_2}) = \frac{c_1 + (-1)^{m_1} c_2 - (c_1 + (-1)^{m_1} c_2 \mod 4)}{4}. \tag{16}$$

We can say that $A_c$ and $A_m$ are $\mathbb{Z}_2$ characters for $C_4$ and $M$, respectively.

The $q = 0$ and $q = 2$ rows of the LHS spectral sequence at the $E_2$ page can be directly obtained from (15). The $q = 1$ row gives $H^*(D_4, (\mathbb{Z}_2)^2)$, which is a module over the point group cohomology ring $H^*(D_4, \mathbb{Z}_2)$, where both $C_4$ and $M$ permute the two $\mathbb{Z}_2$ factors of $(\mathbb{Z}_2)^2$. It turns out to be

$$H^*(D_4, (\mathbb{Z}_2)^2) = H^*(D_4, \mathbb{Z}_2).[\omega_{01}, \omega_{11}]/((A_c + A_m)\omega_{01}, (A_c + A_m)\omega_{11}), \tag{17}$$

where $\omega_{01}$ and $\omega_{11}$ are two generators (of the module) at degree 0 and 1, respectively. Thus, we can write down the standard LHS spectral sequence for $p4g$ in terms of these generators,

| $q=2$ | $\omega_{02}$ | $A_c\omega_{02}, A_m\omega_{02}$ | $A_c^2\omega_{02}, A_m^2\omega_{02}, B_\alpha\omega_{02}$ | $A_c^3\omega_{02}, A_m^3\omega_{02}, B_\alpha A_c\omega_{02}, B_\alpha A_m\omega_{02}$ | $\cdots$ |
|---|---|---|---|---|---|
| $q=1$ | $\omega_{01}$ | $A_m\omega_{01}, \omega_{11}$ | $A_m^2\omega_{01}, B_\alpha\omega_{01}, A_m\omega_{11}$ | $A_m^3\omega_{01}, B_\alpha A_m\omega_{01}, A_m^2\omega_{11}, B_\alpha\omega_{11}$ | $\cdots$ |
| $q=0$ | $1$ | $A_c, A_m$ | $A_c^2, A_m^2, B_\alpha$ | $A_c^3, A_m^3, B_\alpha A_c, B_\alpha A_m$ | $\cdots$ |
| $E_2^{p,q}$ | $p=0$ | $p=1$ | $p=2$ | $p=3$ | $\cdots$ |

$$(18)$$

subject to the relations $(A_c + A_m)\omega_{01} = 0$ and $(A_c + A_m)\omega_{11} = 0$. Also, to account for elements that do not explicitly appear in the spectral sequence, several relations must exist: $\omega_{01}^2 + \cdots = 0$, $\omega_{11}^2 + \cdots = 0$, $\omega_{01}\omega_{11} + \cdots = 0$, $\omega_{02}^2 + \cdots = 0$, $\omega_{02}\omega_{01} + \cdots = 0$, and $\omega_{02}\omega_{11} + \cdots = 0$.

For $p4g$, we have a nontrivial differential $d_2$. From the Leibniz rule Eq. (8), it suffices to write down the image of the generators $\omega_{01}, \omega_{11}$ and $\omega_{02}$ under $d_2$, which are

$$d_2(\omega_{01}) = A_c^2, \quad d_2(\omega_{11}) = 0, \quad d_2(\omega_{02}) = A_m\omega_{11}. \tag{19}$$

Thus, at the $E_3$ page, we have

| $q=0$ | $0$ | $\widetilde{\omega_{12}}$ | $A_m\widetilde{\omega_{12}}$ | $A_m^2\widetilde{\omega_{12}}, B_\alpha\widetilde{\omega_{12}}$ | $\cdots$ |
|---|---|---|---|---|---|
| $q=1$ | $0$ | $\omega_{11}$ | $0$ | $B_\alpha\omega_{11}$ | $\cdots$ |
| $q=0$ | $1$ | $A_c, A_m$ | $A_m^2, B_\alpha$ | $A_m^3, B_\alpha A_c, B_\alpha A_m$ | $\cdots$ |
| $E_3^{p,q}$ | $p=0$ | $p=1$ | $p=2$ | $p=3$ | $\cdots$ |

$$(20)$$

Note that even though $\omega_{01}$ and $\omega_{02}$ have nontrivial image under $d_2$ and are hence killed at the $E_3$ page, not everything involving $\omega_{01}$ or $\omega_{02}$ is killed. Specifically, $d_2((A_c + A_m)\omega_{02}) = (A_c + A_m)A_m \cdot \omega_{11} = 0$ hence $(A_c + A_m)\omega_{02}$ survives and descends to $\widetilde{\omega_{12}}$. It turns out that $E_3$ collapses to $E_\infty$ and the analysis is done. From this, we can directly see that there are five generators in total, i.e., $A_c, A_m$ at degree $p + q = 1$, $\omega_{11}, B_\alpha$ at degree $p + q = 2$ and $\widetilde{\omega_{12}}$ at degree $p + q = 3$. These five generators exactly correspond to the five generators of the mod-2 cohomology ring of $p4g$. Moreover, there are relations involving $A_c^2, A_cA_m, A_c\omega_{11}, A_m\omega_{11}, \omega_{11}^2, A_c\widetilde{\omega_{12}}, \omega_{11}\widetilde{\omega_{12}}, \widetilde{\omega_{12}}^2$ in $E_\infty^{*,*}$, all of which should correspond to an independent relation of the mod-2 cohomology ring of $p4g$.

Indeed, from our code, the mod-2 cohomology ring of $p4g$ is given by

$$\mathbb{Z}_2\left[A_c, A_m, B_\alpha, B_\beta, C_\gamma\right]/\langle\mathcal{R}_2, \mathcal{R}_3, \mathcal{R}_4, \mathcal{R}_5, \mathcal{R}_6\rangle, \tag{21}$$

where the relations $\mathcal{R}_{2,3,4,5,6}$ at degree $2, 3, 4, 5, 6$ are

$$\mathcal{R}_2: \quad A_c^2, \quad A_cA_m, \tag{22a}$$
$$\mathcal{R}_3: \quad A_cB_\beta, \quad A_m(B_\alpha + B_\beta), \tag{22b}$$
$$\mathcal{R}_4: \quad B_\beta(B_\alpha + B_\beta), \quad A_cC_\gamma, \tag{22c}$$
$$\mathcal{R}_5: \quad (B_\alpha + B_\beta)C_\gamma, \tag{22d}$$
$$\mathcal{R}_6: \quad C_\gamma^2 + B_\beta^3 + A_mB_\alpha C_\gamma. \tag{22e}$$

We see that all the generators and relations match our analysis from the LHS spectral sequence. In particular, there are indeed five generators in total, with $A_c, A_m, B_\alpha, B_\beta, C_\gamma$ [4] descending to $A_c, A_m, B_\alpha, \omega_{11}, \widetilde{\omega_{12}}$. The relations all match our analysis of the LHS spectral sequence, even though we cannot fix their explicit form solely from this analysis. Also, we see how a degree-3 generator of the mod-2 cohomology ring of wallpaper groups can appear.

According to the analysis of all 17 wallpaper groups in Ref. [15], it turns out that $p4g$ is the *only* wallpaper group which has a degree-3 generator, while other wallpaper groups have at most degree-2 generators.

### 2. Example: No. 42 (Fmm2) and collapse at $E_2$ page

The analysis for 3D space groups is much more complicated. Still, we manage to perform the analysis for several nontrivial groups. As a side quest, we ask the following question: *when does the standard LHS spectral sequence collapse at $E_2$ page?* This question was raised for general crystallographic groups in any dimension at the end of Ref. [41]. A non-collapsing $E_2$, or equivalently the existence of nonzero differentials $d_2$, can have important physical implications [42, 43].

---

[4] In Ref. [15], they are denoted as $A_c, A_s, B_{c^2}, B_{c(x+y)}, C_{c^2(x+y)}$, respectively. Here we follow the naming convention of 3D space groups in Sec. IV B.

For this purpose, let us study the group No. 42 ($Fmm2$) in more detail. The point group of No. 42 is $C_{2v}$, which as an abstract group is $\mathbb{Z}_2 \times \mathbb{Z}_2$. This group is generated by three translations $T_{1,2,3}$, a two-fold rotation $C_2$, and a mirror reflection $M$:

$$T_1: (x,y,z) \to (x, y+1/2, z+1/2), \tag{23a}$$
$$T_2: (x,y,z) \to (x+1/2, y, z+1/2), \tag{23b}$$
$$T_3: (x,y,z) \to (x+1/2, y+1/2, z), \tag{23c}$$
$$C_2: (x,y,z) \to (-x, -y, z), \tag{23d}$$
$$M: (x,y,z) \to (x, -y, z). \tag{23e}$$

This group is a symmorphic space group which splits, i.e., $G = T \rtimes P$. Yet, we will see that the $E_2$ page of $G$ does not collapse, i.e. $E_2$ page is not isomorphic to $E_\infty$.

The mod-2 cohomology ring for $C_{2v} \cong \mathbb{Z}_2 \times \mathbb{Z}_2$ is

$$H^*(C_{2v}, \mathbb{Z}_2) = \mathbb{Z}_2[A_c, A_m], \tag{24}$$

where the generators $A_c$ and $A_m$ have the following explicit cochain representative in the bar resolution,

$$A_c(C_2^c M^m) = c, \quad A_m(C_2^c M^m) = m, \quad c, m \in \{0, 1\}. \tag{25}$$

We can say that $A_c$ and $A_m$ are $\mathbb{Z}_2$ characters for $C_2$ and $M$, respectively.

The $q = 0$ and $q = 3$ rows of the $E_2$ page can be directly obtained from Eq. (24). For the $q = 1$ row, we have

$$H^*(C_{2v}, H^1(\mathbb{Z}^3, \mathbb{Z}_2)) = H^*(C_{2v}, \mathbb{Z}_2).[\omega_{01}, \omega'_{01}]/(A_c\omega_{01} + A_m\omega'_{01}), \tag{26}$$

where $\omega_{01}$ and $\omega'_{01}$ are two generators (of the module) at degree 0; For the $q = 2$ row, we have

$$H^*(C_{2v}, H^2(\mathbb{Z}^3, \mathbb{Z}_2)) = H^*(C_{2v}, \mathbb{Z}_2).[\omega_{02}, \omega_{12}]/(A_c\omega_{02}, A_m\omega_{02}). \tag{27}$$

where $\omega_{02}$ and $\omega_{12}$ are two generators at degree 0 and 1, respectively. Therefore, the LHS spectral sequence at the $E_2$ page has the following form

| $q=3$ | $\omega_{03}$ | $A_c\omega_{03}, A_m\omega_{03}$ | $A_c^2\omega_{03}, A_cA_m\omega_{03}, A_m^2\omega_{03}$ | $A_c^3\omega_{03}, A_c^2A_m\omega_{03}, A_cA_m^2\omega_{03}, A_m^3\omega_{03}$ | $\cdots$ |
|---|---|---|---|---|---|
| $q=2$ | $\omega_{02}$ | $\omega_{12}$ | $A_c\omega_{12}, A_m\omega_{12}$ | $A_c^2\omega_{12}, A_cA_m\omega_{12}, A_m^2\omega_{12}$ | $\cdots$ |
| $q=1$ | $\omega_{01}, \omega'_{01}$ | $A_c\omega_{01}, A_m\omega_{01}, A_c\omega'_{01}$ | $A_c^2\omega_{01}, A_cA_m\omega_{01}, A_m^2\omega_{01}, A_c^2\omega'_{01}$ | $A_c^3\omega_{01}, A_c^2A_m\omega_{01}, A_cA_m^2\omega_{01}, A_m^3\omega_{01}, A_c^3\omega'_{01}$ | $\cdots$ |
| $q=0$ | $1$ | $A_c, A_m$ | $A_c^2, A_cA_m, A_m^2$ | $A_c^3, A_c^2A_m, A_cA_m^2, A_m^3$ | $\cdots$ |
| $E_2^{p,q}$ | $p=0$ | $p=1$ | $p=2$ | $p=3$ | $\cdots$ |

$$\tag{28}$$

It turns out that $d_2$ maps $\omega_{01}$, $\omega'_{01}$, $\omega_{02}$, and $\omega_{03}$ to zero while we calculate in Appendix D that it maps $\omega_{12}$ to

$$d_2(\omega_{12}) = (A_c^3 + A_cA_m^2)\omega_{01}. \tag{29}$$

Hence, the $E_2$ page is non-collapsing for the symmorphic group No. 42 ($Fmm2$). At the $E_3$ page, we have

| $q=3$ | $\omega_{03}$ | $A_c\omega_{03}, A_m\omega_{03}$ | $A_c^2\omega_{03}, A_cA_m\omega_{03}, A_m^2\omega_{03}$ | $A_c^3\omega_{03}, A_c^2A_m\omega_{03}, A_cA_m^2\omega_{03}, A_m^3\omega_{03}$ | $\cdots$ | |
|---|---|---|---|---|---|---|
| $q=2$ | $\omega_{02}$ | $0$ | $0$ | $0$ | $\cdots$ | |
| $q=1$ | $\omega_{01}, \omega'_{01}$ | $A_c\omega_{01}, A_m\omega_{01}, A_c\omega'_{01}$ | $A_c^2\omega_{01}, A_cA_m\omega_{01}, A_m^2\omega_{01}, A_c^2\omega'_{01}$ | $A_c^3\omega_{01}, A_c^2A_m\omega_{01}, A_m^3\omega_{01}, A_c^3\omega'_{01}$ | $\cdots$ | (30) |
| $q=0$ | $1$ | $A_c, A_m$ | $A_c^2, A_cA_m, A_m^2$ | $A_c^3, A_c^2A_m, A_cA_m^2, A_m^3$ | $\cdots$ | |
| $E_2^{p,q}$ | $p=0$ | $p=1$ | $p=2$ | $p=3$ | $\cdots$ | |

$E_3$ collapses to $E_\infty$ and the analysis is done. Also, we have $\omega_{02} = \omega_{01}\omega'_{01}$. From this, we can directly see that there are five generators in total, i.e., $A_c, A_m, \omega_{01}, \omega'_{01}$ at degree $p+q = 1$ and $\omega_{03}$ at degree $p+q = 3$, and we should have relations involving $A_c\omega_{01} + A_m\omega'_{01}$, $\omega_{01}^2$, $\omega'^2_{01}$, $(A_c^3 + A_cA_m^2)\omega_{01}$, $\omega_{01}\omega_{03}$, $\omega'_{01}\omega_{03}$ and $\omega_{03}^2$.

The $\mathbb{Z}_2$ cohomology ring of No. 42 ($Fmm2$) is given by

$$\mathbb{Z}_2[A_c, A_m, A_{x+z}, A_{y+z}, C_{xyz}]/\langle \mathcal{R}_2, \mathcal{R}_4, \mathcal{R}_6 \rangle \tag{31}$$

where the relations are

$$\mathcal{R}_2: \quad A_cA_{x+z} + A_mA_{x+z} + A_cA_{y+z}, \quad A_{x+z}^2 + A_cA_{y+z}, \quad A_{y+z}(A_c + A_{y+z}), \tag{32a}$$
$$\mathcal{R}_4: \quad A_{x+z}C_{xyz}, \quad A_{y+z}C_{xyz}, \tag{32b}$$
$$\mathcal{R}_6: \quad C_{xyz}^2. \tag{32c}$$

There are indeed five generators in total, with $A_c, A_m, A_{x+z} + A_{y+z}, A_{x+z}, C_{xyz}$ descending to $A_c, A_m, \omega_{01}, \omega'_{01}, \omega_{03}$. Note that the relation $(A_c^3 + A_c A_m^2)(A_{x+z} + A_{y+z})$ is not an independent relation and can be derived from the relations $\mathcal{R}_2$. Other relations match the relations in $E_\infty^{*,*}$.

A number of examples of split crystallographic groups whose LHS spectral sequence does not collapse at $E_2$ were given in Ref. [44], and later the obstruction characteristic classes were studied in Ref. [45]. Here we provide a complete list for this phenomenon to occur in 3D space groups. Out of the 73 symmorphic space groups, the following 10 symmorphic groups do not have a collapsing $E_2$ page:

$$
\begin{aligned}
&42\ (Fmm2), \quad 69\ (Fmmm), \quad 87\ (I4/m), \quad 107\ (I4mm), \quad 121\ (I\bar{4}2m), \\
&139\ (I4/mmm), \quad 202\ (Fm\bar{3}), \quad 217\ (I\bar{4}3m), \quad 225\ (Fm\bar{3}m), \quad 229\ (Im\bar{3}m).
\end{aligned}
\tag{33}
$$

Note the subgroup relations

$$
\begin{aligned}
&42 \subset 69, \quad 69, 87, 107, 121 \subset 139, \quad 121 \subset 217, \\
&69 \subset 202, \quad 139, 202 \subset 225, \quad 139, 217 \subset 229,
\end{aligned}
$$

therefore, the issue of a nonzero differential $d_2$ starts to appear at No. 42 ($Fmm2$) [5], and this propagates to the larger symmorphic groups containing No. 42 as a subgroup. Moreover, the following 13 nonsymmorphic groups have a collapsing $E_2$ page:

$$
\begin{aligned}
&77\ (P4_2), \quad 93\ (P4_222), \quad 144\ (P3_1), \quad 145\ (P3_2), \quad 151\ (P3_112), \quad 152\ (P3_121), \quad 153\ (P3_212), \\
&154\ (P3_221), \quad 171\ (P6_2), \quad 172\ (P6_4), \quad 180\ (P6_222), \quad 181\ (P6_422), \quad 208\ (P4_232).
\end{aligned}
\tag{34}
$$

These groups and the 63 (of the 73) symmorphic groups that have a collapsing $E_2$ page are labeled with a "⬥" in the third column of Table IV. The 13 nonsymmorphic groups in Eq. (34) and the 73 symmorphic groups constitute the 86 groups having $E_\infty^{0,3} = \mathbb{Z}_2$ in the standard LHS spectral sequence. All other 144 groups have $E_2^{0,3} = \mathbb{Z}_2$ and $E_3^{0,3} = 0$.

Finally, a refined conjecture was raised by Adem, Ge, Pan, and Petrosyan [41]: the $E_2$ page collapses for $k$-dimensional crystallographic groups $G = \mathbb{Z}^k \rtimes P$ when $P$ is a cyclic group. Here we find that this conjecture holds for all the 3D ($k = 3$) symmorphic space groups of this form. A counterexample has been found in the 6D ($k = 6$) crystallographic groups [46].

## B. Finiteness theorem

Building on the discussion regarding the LHS spectral sequence, in this subsection, we prove the following theorem, which states that the mod-2 cohomology ring of any crystallographic group $G$ is finitely generated.

**Theorem 4.** *(Finiteness condition) The $\mathbb{F}$ cohomology ring of any crystallographic group $G$ must be finitely generated with a finite number of generators and relations.*

The proof follows the corresponding proof for finite groups in Ref. [37], also based on the analysis of the LHS spectral sequence. This theorem guarantees that our code always ends at a finite degree.

*Proof.* Consider the standard LHS spectral sequence. At the $E_2$ page, every row is of the form $H^*(P, \mathbb{F}^m)$ where $\mathbb{F}^m$ is a finitely generated $\mathbb{F}$-module. The following theorem dictates that every row is a finitely generated module, i.e., a Noetherian module, over $H^*(P, \mathbb{F})$ [6].

**Theorem 5.** *(Evens [37]) Let $P$ be a finite group, $k$ a commutative ring on which $P$ acts trivially, and $M$ a $kP$-module. If $M$ is Noetherian as a $k$-module, then $H^*(P, M)$ is Noetherian over $H^*(P, k)$.*

Together with the bilinear product at the $E_2$ page, we see that the $E_2$ page of the standard LHS spectral sequence is a Noetherian module over $H^*(P, \mathbb{F})$. Starting from the $E_2$ page, the $E_{r+1}$ page is a quotient module of a submodule, i.e., $\ker d_r / \operatorname{im} d_r$, of the $E_r$ page. Since the submodule and the quotient module of a Noetherian module are all Noetherian, the $E_r$ page for any $r$ is Noetherian. Corollary 3 says that $E_{k+2} = E_\infty$, hence the $E_\infty$ page is also a Noetherian module over $H^*(P, \mathbb{F})$. The same proof as in Lemma 7.4.5 of Ref. [37] then dictates that $H^*(G, \mathbb{F})$ is a Noetherian module over $H^*(P, \mathbb{F})$. Therefore, the Hilbert basis theorem dictates that $H^*(G, \mathbb{F})$ is a Noetherian ring itself. By the usual argument, the cohomology in positive degrees, denoted as $H^+(G, \mathbb{F})$, is a finitely generated ideal, and a set of ideal generators is also a set of generators for $H^*(G, \mathbb{F})$ as an algebra over $\mathbb{F}$. The Noetherian condition also guarantees that we only need a finite number of relations.

$\square$

---

[5] Note that this group is an example for Corollary 1 of Ref. [44].

[6] In general, a module is said to be *Noetherian* if every submodule is finitely generated. Still, we know that $H^*(P, \mathbb{F})$ is a Noetherian ring, and a finitely generated module over a Noetherian ring is a Noetherian module.

## C. On the integral cohomology of crystallographic groups

In this subsection, we discuss the relevant maps to relate mod-2 cohomology with integral cohomology. This connects our results of mod-2 cohomology with cohomology with other coefficients. The complete results for the integral cohomology of the 230 space groups are collected in Table V.

Specifically, from the short exact sequence

$$\mathbb{Z} \xrightarrow{i=\times 2} \mathbb{Z} \xrightarrow{p=\mathrm{mod}\ 2} \mathbb{Z}_2, \tag{35}$$

we have the long exact sequence

$$\cdots \to H^{n-1}(G,\mathbb{Z}_2) \xrightarrow{\beta_{n-1}} H^n_\rho(G,\mathbb{Z}) \xrightarrow{i_n^*=\times 2} H^n_\rho(G,\mathbb{Z}) \xrightarrow{p_n^*=\mathrm{mod}\ 2} H^n(G,\mathbb{Z}_2) \xrightarrow{\beta_n} \cdots \tag{36}$$

where $i_n^*$ and $p_n^*$ are maps induced by the homomorphism $i\colon \mathbb{Z} \to \mathbb{Z}$ and $p\colon \mathbb{Z} \to \mathbb{Z}_2$ at degree $n$, and $\beta_n$ is the Bockstein homomorphism at degree $n$. Here we allow any action $\rho$ of $G$ on the module $\mathbb{Z}$.

From this we deduce that there is a (not necessarily natural) decomposition of the group $H^n(G,\mathbb{Z}_2)$

$$H^n(G,\mathbb{Z}_2) = (H^n_\rho(G,\mathbb{Z}) \otimes \mathbb{Z}_2) \bigoplus \mathrm{Tor}^{\mathbb{Z}}(H^{n+1}_\rho(G,\mathbb{Z}),\mathbb{Z}_2). \tag{37}$$

This is just a form of Universal Coefficient Theorem for $\mathbb{Z}_2$ (which holds for any action $\rho$ of $G$ on $\mathbb{Z}$).

It turns out that the cohomology elements corresponding to the LSM constraints are always contained in the first summand $H^n_\rho(G,\mathbb{Z}) \otimes \mathbb{Z}_2$ (see Statement 16 and 18), where the action $\rho$ of $G$ on the module $\mathbb{Z}$ is a particular one dictated by the Crystalline Equivalence Principle [20], such that reflection and inversion symmetries act on $\mathbb{Z}$ by $-1$. We will call this action the *orientation-reversing* action and the associated $\mathbb{Z}$ module the orientation-reversing $\mathbb{Z}$ module, and we denote the $\mathbb{Z}$ module by $\mathbb{Z}^{\mathrm{or}}$. To characterize this subset in another way, let us introduce the *twisted* Steenrod operation $\mathcal{SQ}^1$ at degree $n$,

$$\mathcal{SQ}^1 := p_{n+1}^* \circ i_{n+1}^* \circ \beta_n \colon H^n(G,\mathbb{Z}_2) \to H^{n+1}(G,\mathbb{Z}_2). \tag{38}$$

From the long exact sequence Eq. (36), we see that the first summand $H^n(G,\mathbb{Z}^{\mathrm{or}}) \otimes \mathbb{Z}_2$ is exactly the kernel of $\mathcal{SQ}^1$ in $H^n(G,\mathbb{Z}_2)$.

We call $\mathcal{SQ}^1$ the twisted Steenrod operation for the following reasons. First of all, when the action of $G$ on $\mathbb{Z}$ is trivial, $\mathcal{SQ}^1$ is exactly the original Steenrod operations. Secondly, elementary calculations [15] show that for $\lambda \in H^n(G,\mathbb{Z}_2)$, we have

$$\mathcal{SQ}^1(\lambda) = Sq^1(\lambda) + \mathcal{SQ}^1(1) \cup \lambda, \tag{39}$$

where $1 \in H^0(G,\mathbb{Z}_2) \cong \mathbb{Z}_2$. Here, $\mathcal{SQ}^1(1) \in H^1(G,\mathbb{Z}_2)$ can be thought of as a map $G \to \mathbb{Z}_2$ which maps the element $g \in G$ to the nontrivial $1 \in \mathbb{Z}_2$ (trivial $0 \in \mathbb{Z}_2$) if $g$ acts nontrivially (trivially) on the $\mathbb{Z}$ module. We will return to this operation in Sec. V when we discuss LSM constraints.

## D. Periodicity of mod-2 cohomology

In this subsection, we collect some interesting theorems about the periodicity of $\mathbb{Z}_2$ rank for mod-2 cohomology of crystallographic groups. Our results for 2D wallpaper groups and 3D space groups indeed align with these general theorem applicable to all crystallographic groups.

We have the following theorem regarding the periodicity of the resolution for a group $G$ [7]

**Theorem 6.** *(Brown [40]) A group $G$ with a finite index torsion free nilpotent subgroup admits a resolution which is periodic in sufficiently high degrees if and only if all of its finite index subgroups admit periodic resolutions.*

Therefore, if we want the $\mathbb{Z}_2$ ranks for mod-2 cohomology of crystallographic groups to be periodic in sufficiently high degrees, we need to check all of its finite index subgroups.

An interesting class of crystallographic groups is called Bieberbach group [5], which is the extreme limit of crystallographic groups which satisfy the criterion of Theorem 6.

**Definition 4.** *(Bieberbach group) A crystallographic group is a Bieberbach group if it is torsion free.*

———

[7] We always assume that the resolution is with respect to $\mathbb{Z}$.

As such, a Bieberbach group satisfies the criterion of Theorem 6 and hence admits a periodic resolution in sufficiently high degree.

In fact we can say more about these Bieberbach groups [5]. For a $k$-dimensional crystallographic group $G$, consider the orbit space $\mathbb{R}^k/G$. For generic crystallographic group $G$, $\mathbb{R}^k/G$ may contain some singularities. However, when $G$ is a Bieberbach group, $\mathbb{R}^k/G$ is a *smooth* manifold. Even more, $\mathbb{R}^k/G$ is a $k$-dimensional *flat* manifold, which is a compact closed Riemannian manifold with zero sectional curvature. On the other hand, any $k$-dimensional flat manifold can be considered as an orbit space $\mathbb{R}^k/G$ for some Bieberbach group $G$. Moreover, we can immediately see that $G$ is precisely the fundamental group of $\mathbb{R}^k/G$, and $\mathbb{R}^k/G$ is in fact a $K(G,1)$, i.e., the classifying space of $G$.

This correspondence allows us to establish a one-to-one correspondence with $k$-dimensional flat manifolds with $k$-dimensional Bieberbach groups. Moreover, the group cohomology of $G$ should be isomorphic to the simplicial cohomology of $K(G,1)$, which we see is a $k$-dimensional smooth manifold. Hence, we establish the following theorem regarding Bieberbach groups.

**Theorem 7.** *A $k$-dimensional Bieberbach group $G$ satisfies the condition that $H^k(G,\mathbb{Z}) \cong \mathbb{Z}$ and $H^n(G,\mathbb{Z}) = 0, n > k$.*

Similarly, regarding mod-2 cohomology, we also have $H^k(G,\mathbb{Z}_2) \cong \mathbb{Z}_2$ and $H^k(G,\mathbb{Z}) = 0, n > k$. We will see from specific examples of 2D wallpaper groups and 3D space groups that this is indeed the case.

From the point of view of $k$-dimensional lattice with symmetry $G$, $\mathbb{R}^k/G = K(G,1)$ can be thought of as the *fundamental domain* associated with crystallographic symmetry $G$, which also corresponds to the unique IWP for the Bieberbach group $G$.

The criterion of Theorem 6 can also be satisfied if the point group $P$ of $G$ admits periodic resolution. Hence, we have the following easy corollary (see also Ref. [9] page 247–248),

**Corollary 8.** *For an $k$-dimensional crystallographic group $G$ with the associated point group $P$, suppose that the point group $P$ admits a periodic resolution of period $d$. Then, the crystallographic group $G$ admits a periodic resolution of period $d$ in degrees greater than $n$. In particular, $H^k(G,\mathbb{Z}_2) = H^{k+d}(G,\mathbb{Z}_2)$ for all integers $k > n$.*

### 1. Example: 2D wallpaper groups

To illustrate these theorems, let us check the criterion for 2D wallpaper groups and we will see that these theorems indeed match with the explicit calculation for individual groups in Ref. [15].

1. There are exactly two Bieberbach groups in 2D, $p1$ and $pg$. For both groups $G$, we have $H^2(G,\mathbb{Z}_2) = \mathbb{Z}_2$ and $H^k(G,\mathbb{Z}_2) = 0, k \geq 3$. The flat manifolds $\mathbb{R}^2/G = K(G,1)$ for $p1$ and $pg$ are in fact torus and the Klein bottle, respectively.

2. For 2D wallpaper groups, there are six point groups satisfying the condition of having a periodic resolution:

$$C_2, D_1 \cong \mathbb{Z}_2, \quad C_4 \cong \mathbb{Z}_4, \quad C_3 \cong \mathbb{Z}_3, \quad D_3 \cong Dih_3, \quad C_6 \cong \mathbb{Z}_6. \tag{40}$$

The period is one for $\mathbb{Z}_2$, two for $\mathbb{Z}_4$, $\mathbb{Z}_3$, $\mathbb{Z}_6$, and four for $Dih_3$, and for each of these groups, the mod-2 cohomology groups in all positive degrees are the same [40, 47]. The corresponding wallpaper groups are $p2$ ($C_2$), $pm$, $pg$, $cm$ ($D_1$), $p4$ ($C_4$), $p3$ ($C_3$), $p3m1$, $p31m$ ($D_3$) and $p6$ ($C_6$). We see that indeed for each of these wallpaper groups, the mod-2 cohomology groups in degrees greater than 2 are the same.

3. There are two more wallpaper groups—$pmg$ and $pgg$—that satisfy the condition of Theorem 6. They also have a periodic resolution and the mod-2 cohomology groups in degrees equal to or greater than 2 are the same as well.

We will see that these patterns reappear in the study of 3D space groups.

## IV. Mod-2 Cohomology of 3D space groups

In this section, we sketch how we perform the calculation for the cohomology of crystallographic groups in 2D and 3D, and highlight certain important features of the results.

There are standard textbook methods for calculating group cohomology [38, 40], many of which are already implemented in GAP [9]. The GAP routine of computing cohomology rings has so far been limited to $p$-groups. Nevertheless, one can use the elementary GAP routines as building blocks and calculate the mod-2 cohomology ring of the space group cohomology $H^*(G,\mathbb{Z}_2)$ in an algorithmic way. We follow this strategy and create a program tailored for computing the mod-2 cohomology ring of space groups [8]. We hope our algorithm will eventually be useful for calculating the cohomology rings of more general groups.

--------

[8] The GAP code scripts for computing the mod-2 cohomology ring of space groups, `SpaceGroupCohomologyData.gi` and `SpaceGroupCohomologyFunctions.gi`, can be found in our Github repository [26].

Even more explicitly, to connect these cohomology data to lattice data, we managed to write down the inhomogeneous functions representing the cohomology generators at degree-1, 2, and 3. We write additional codes to obtain the mapping from these function-represented cocycles to the vector-represented cocycles, connecting the output data of these two separate methods. The explicit inhomogeneous functions allows us to evaluate topological invariants associated with IWPs, establishing the final link we aim for—a correspondence between 3-cocycles and IWPs.

## A. Methods of calculation

The datum encoding the cohomology of any group $G$ is a *free $\mathbb{Z}G$ resolution of $\mathbb{Z}$*: it is an acyclic chain complex consisting of free $\mathbb{Z}G$-modules $R_n^G$ and homomorphisms $\partial_n$

$$R_*^G: \cdots \xrightarrow{\partial_{n+2}} R_{n+1}^G \xrightarrow{\partial_{n+1}} R_k^G \xrightarrow{\partial_n} R_{n-1}^G \xrightarrow{\partial_{n-1}} \cdots \xrightarrow{\partial_2} R_1^G \xrightarrow{\partial_1} R_0^G \xrightarrow{\partial_0} \mathbb{Z}, \tag{41}$$

such that $\ker \partial_n = \operatorname{im} \partial_{n+1}$ for every $n$. The cohomology calculation we did relies exclusively on the construction and manipulation of the resolution.

The resolution $R_*^G$ may be finitely or infinitely generated, depending on whether each $R_k^G$ is finitely or infinitely generated as a $\mathbb{Z}G$-module. For an infinite-order crystallographic group $G$, GAP always produces a finitely generated resolution, which is easy to work with but quite abstract. In contrast, the familiar bar resolution provides clearer meanings but is infinitely generated, making it more challenging to use. These two resolutions serve as the foundation for the two independent calculations presented in Secs. IV A 1 and IV A 2.

In Sec. IV A 1, we outline the standard procedures implemented in GAP for computing group cohomology and highlight the algorithm that we develop in order to obtain the ring structure for crystallographic groups. Then, in Sec. IV A 2, we explain how the familiar bar resolution method can be applied to obtain the standard cocycle functions for crystallographic groups. The key to this calculation is a conjecture on the restriction of cohomology to a finite lattice space group that, if holds, allows to convert the computation of cocycle functions from an infinite problem to a finite problem. Finally, in Sec. IV A 3 we explain how to connect the outputs of the two methods using the contracting homotopy for the bar resolution.

### 1. Using finite-dimensional free resolution from GAP

This computation leverages the full power of GAP [8]. The resolution (41) is computed internally in GAP. Once the resolution is constructed, the cohomology $H^n(G, A)$ as an abelian group is then obtained in a straightforward manner by first taking the Hom functor with the coefficient module $A$ and then the homology of the cochain maps. The calculation of the ring structure in GAP relies crucially on the data of *contracting homotopy*:

**Definition 5.** *(Contracting homotopy. [9]) Let $R_*^G$ be a free $\mathbb{Z}G$-resolution of $\mathbb{Z}$. A contracting homotopy on $R_*^G$ consists of a sequence of abelian group homomorphisms $h_n: R_n^G \to R_{n+1}^G, n \geq 0$, satisfying*

$$h_{n-1}\partial_n + \partial_{n+1}h_n = 1, \tag{42}$$

*for $n \geq 0$ with $h_{-1} = 0$. The homomorphisms $h_k$ do not have to preserve the action of $G$.*

For $G$ being a crystallographic group, contracting homotopy up to high degrees is produced only via the following GAP command

$$\texttt{ResolutionAlmostCrystalGroup}. \tag{43}$$

The data of resolution and contracting homotopy make the evaluation of cup products a routine task. However, significant limitations exist in obtaining the full mod-2 cohomology ring of space groups in GAP, necessitating the development of a specialized program:

- The command (43) suffers from memory constraints, and is only capable of producing a resolution $R_{\leq 6}^G$ up to degree six for space groups No. 221–230 [9] . This prevents us from obtaining the complete mod-2 cohomology ring of space groups No. 226, 228, and 230 which contain relations at degree 7 or higher.

- Computing cup products can be highly time-consuming. For example, the computation of the degree-8 relation for space group No. 142 can take several days. The bottleneck lies in computing the composition of contracting homotopy.

———————

[9] To our knowledge, there is no way to obtain the degree-7 resolution (equipped with a contracting homotopy) for space groups No. 221–230. This is essentially due to the extremely high memory cost in computing the resolution for point group $O_h$ within the command `ResolutionFiniteGroup` which is called internally by (43).

We reprogram the part of calculating the cup products in a way that alleviates the problems mentioned above. In our code, when computing the cup product between $u$ and several elements $v_1, v_2, ...$, we optimize by computing the (composition of) contracting homotopy for $u$ only once. When computing the cup product of two cocycles $u$ and $v$, we always choose the one such that fewer compositions of contracting homotopy need to be calculated. Finally, the computation of cup products proceeds from lower to higher degrees, and we make use of the ring relations already obtained at lower degrees to avoid evaluating unnecessary cup products at higher degrees.

Once cup products are computed, we use linear algebra methods to obtain a presentation of the mod-2 cohomology ring by choosing a minimal set of generators and relations. The GAP resolution $R_*^G$ carries information about the standard LHS spectral sequence (to be introduced in Sec. III A), and we use this information to label the mod-2 cohomology ring generators (see Sec. IV B for the labeling convention).

### 2. Using bar resolution

Independent of the above method and codes, for a given space group $G$, we also sought to write down representative inhomogeneous functions (explicit cochain expressions)

$$f \colon \underbrace{G \times \cdots \times G}_{n \text{ times}} \to \mathbb{Z}_2 \tag{44}$$

for the mod-2 cohomology ring generators of degree $n$ equal to or less than three ($n \leq 3$). To find such a cochain function $f$, we solve the cocycle condition

$$(df)(g_1, g_2, ..., g_{n+1}) := f(g_2, ..., g_{n+1}) + \sum_{j=1}^{n}(-1)^j f(g_1, ..., g_j g_{j+1}, ..., g_n) + (-1)^{n+1} f(g_1, ..., g_n) = 0 \tag{45}$$

for all $g_1, g_2, ..., g_{n+1} \in G$. The resolution underlying Eq. (45) is the bar resolution: for $n \geq 0$, each $R_n^G$ in Eq. (41) is the free $\mathbb{Z}G$-module freely generated by $n$-tuples $[g_1|g_2|\cdots|g_n]$ with $g_i \in G$, and the boundary map is defined by

$$\partial_n[g_1|\cdots|g_n] = g_1[g_2|\cdots|g_n] + \sum_{i=1}^{n-1}(-1)^i[g_1|\cdots|g_i g_{i+1}|\cdots|g_n] + (-1)^n[g_1|\cdots|g_{n-1}]. \tag{46}$$

Since the space group $G$ has infinite order, Eq. (45) is a system of infinite number of equations and solving them seems to be a hopeless task. Nevertheless, the following conjecture allows us to convert the problem to a finite dimensional problem. To state this conjecture, let us define the "translating-by-$m$-units" subgroup of the translation group: $T^m = (m\mathbb{Z})^3 \subset T \cong \mathbb{Z}^3$. Obviously, $T^m$ is a normal subgroup of the space group $G$, and this defines a quotient group $P_m$ through

$$T^m \to G \xrightarrow{p_m} P_m. \tag{47}$$

The quotient group $P_m$ can be viewed as the symmetry group of a finite lattice that spans $m \times m \times m$ unit cells along the translation direction $T_1$, $T_2$, and $T_3$.

**Conjecture 9.** *When $m = 4$, the induced map $p_m^* \colon H^n(P_m, \mathbb{Z}_2) \to H^n(G, \mathbb{Z}_2)$ is surjective for all $n$.*

This is equivalent to the conjecture that the LHS spectral sequence associated with Eq. (47) collapses to the bottom horizontal line $E_r^{p,0}$ at infinity page $r = \infty$.

For a given space group $G$, we first write down an ansatz for the cocycle function (44). Then we solve the cocycle condition (45) by restricting elements $g_1, g_2, ..., g_{n+1}$ to the group $P_m$. As $P_m$ is a finite group, a cocycle

$$\bar{f} \colon \underbrace{P_m \times \cdots P_m}_{n \text{ times}} \to \mathbb{Z}_2 \tag{48}$$

can be solved (at least in principle). Then, Conjecture 9, if holds, allows us to pull back the cocycle $[\bar{f}] \in H^n(P_m, \mathbb{Z}_2)$ to $[p_m^*(\bar{f})] \in H^n(G, \mathbb{Z}_2)$ hence obtain a cocycle of the space group $G$. By following this strategy we have successfully obtained all the 1-, 2-, and 3-cocycles (for $n = 1, 2, 3$) for 227 of the 230 space groups [10], confirming the validity of the conjecture in these cases. The explicit expression for these cochain functions is stored in the Mathematica file `Space_Group_Cohomology_Data.nb` available on our Github repository [26]. They are labeled by the same name as the ring generators obtained from the GAP program and are the representative inhomogeneous functions of them. This way we achieved at a clear and transparent characterization of the mod-2 cohomology ring.

Compared with the vector-represented cocycles, the explicit function-represented cocycles have several merits:

---

[10] Although working in principle, in practice this strategy is too memory costly for obtaining the degree-3 ring generators of groups No. 225, 227, and 229.

- The explicit cochain functions facilitate the calculation of cup products as the cup product coincides with the usual product of functions;

- They allow to obtain the explicit restriction map associated with the subgroup embedding $H \subset G$. This is extremely useful from a physics point of view in the study of symmetry breaking as it allows us to trace how the LSM anomaly (to be introduced in Sec. V) survives under the lattice symmetry breaking.

- These cochain functions allow us to evaluate the topological invariants at degree $n \leq 3$. Specifically, this allow us to "diagonalize" the deg-3 cohomology elements and find the element that *uniquely* detects the LSM anomaly associated with an IWP, hereby establishing a correspondence between cohomology data and lattice data.

### 3. Connecting the two methods

Finally, in order to establish connection between the bar resolution calculation and the GAP program, we seek to convert a cochain function obtained from the bar resolution to a finite dimensional vector. This can be done using the contracting homotopy of bar resolutions

$$h_n \colon R_n^G \to R_{n+1}^G, \quad g[g_1|\cdots|g_n] \mapsto [g|g_1|\cdots|g_n]. \tag{49}$$

To map a function-represented cocycle (44) to a vector-represented cocycle associated with the free $\mathbb{Z}G$ resolution of $\mathbb{Z}$ given by GAP, $f \mapsto f^*$, one needs to specify how the basis of the latter, $e_i^n$ for $i = 1, 2, ..., b_n$, is mapped to the basis of the bar resolution at the same degree:

$$e_i^n \mapsto \sum [g_1|\cdots|g_n]. \tag{50}$$

The basis map (50) can be built recursively via the contracting homotopy (49) [10]. This is implemented in our GAP code in [26]. Once the basis map is constructued, the vector-represented cocycle is obtained:

$$f^*(e_i^n) = f(\sum [g_1|\cdots|g_n]) = \sum f(g_1, ..., g_n). \tag{51}$$

### B. Mod-2 cohomology ring of 3D space groups

The main result of this work is that we obtain the mod-2 cohomology rings for 229 of the 230 3D space groups [11], which is collected in Appendix F. In this subsection, we show how to read these results and point out some interesting features. We present a mod-2 cohomology ring as follows:

$$H^*(G, \mathbb{Z}_2) = \mathbb{Z}_2[A_\bullet, \ldots, B_\bullet, \ldots, \ldots]/\text{Relations}. \tag{52}$$

Here, $A_\bullet$, $B_\bullet$, ... are the generators of the ring, living in $H^1(G, \mathbb{Z}_2)$, $H^2(G, \mathbb{Z}_2)$, ..., respectively (all the way to $F_\bullet \in H^6(G, \mathbb{Z}_2)$, see Item 1). Subscripts "$\bullet$" give labels to the generators:

- For every space group $G$, we choose a set of group generators, collected for each group in Appendix F (see also Table I for the choice of group generators for each point group). An element $g \in G$ is written in terms of these group generators. For example, No. 1 ($P1$) is generated by three translations $T_{1,2,3}$ along three directions given by Eq. (88), and an element in No. 1 ($P1$) is written as $T_1^x T_2^y T_3^z$ with $x, y, z \in \mathbb{Z}$. More complicatedly, an element in No. 227 ($Fd\bar{3}m$) can be written as $T_1^x T_2^y T_3^z C_2^c C_2'^{c'} C_3^{c_3} M^m I^i$ with $x, y, z \in \mathbb{Z}, c, c', m, i \in \{0, 1\}, c_3 \in \{0, 1, 2\}$.

- A degree-1 generator $A_\bullet \colon G \to \mathbb{Z}_2$ (see Eq. (44)) is labeled by the exponent of group generator that it detects in this decomposition. For example, the degree-1 generator $A_c$, $A_m$ and $A_i$ evaluates to 1, i.e., $A_c(g) = 1$, $A_m(g) = 1$ or $A_i(g) = 1$, whenever the exponenet of a two-fold rotation $C_2$, the reflection $M$ or the inversion $I$ is odd in this decomposition of $g$. Similarly, degree-1 generators $A_{x,y,z}$ detect (the exponents of) translations $T_{1,2,3}$. We will also use labels like $A_{x+y+z}$ (i.e. with "$+$" in the subscript) to indicate that this generator reduces to $A_x + A_y + A_z$ when restricting to the translation subgroup $T$ [12].

---

[11] The only exception being No. 226 ($Fm\bar{3}c$) whose degree-7 and higher relations have not been worked out.

[12] An element $A_{x+y+z}$ is defined by

$$A_{x+y+z}(g) = x + y + z \tag{53}$$

for translation group $T \cong$ No. 1 (P1), but in larger groups the 1-cocycle with the same label may in addition depend on (the exponents of) point group elements. As an example, for group No. 98 ($I4_1 22$), we have

$$A_{x+y+z}(g) := x + y + z + c, \quad g = T_1^x T_2^y T_3^z C_2^c C_2'^{c'} C_2''^{c''} \in \text{No. 98 } (I4_1 22). \tag{54}$$

This notation may result in ambiguities when one tries to write down the restriction $i^* \colon A_{x+y+z} \to A_{x+y+z}$, and one must bear in mind that the $A_{x+y+z}$ on the left-hand side (resp. right-hand side) of the arrow has the expression in Eq. (54) (resp. Eq. (53)). This ambiguity only happens for the following 14 groups

$$
\begin{aligned}
&77 \ (P4_2), \quad 80 \ (I4_1), \quad 93 \ (P4_2 22), \quad 94 \ (P4_2 2_1 2), \quad 98 \ (I4_1 22), \quad 134 \ (P4_2/nnm), \quad 144 \ (P3_1), \\
&151 \ (P3_1 12), \quad 152 \ (P3_1 21), \quad 172 \ (P6_4), \quad 181 \ (P6_4 22), \quad 199 \ (I2_1 3), \quad 206 \ (Ia\bar{3}), \quad 214 \ (I4_1 32).
\end{aligned}
\tag{55}
$$

- We label degree-2 or higher generators according to the entries of the standard LHS spectral sequence (11) that they live in. Specifically, $B_\alpha, B_\beta$ live in the $E_\infty^{2,0}, E_\infty^{1,1}$ entries, and $C_\alpha, C_\beta, C_\gamma$ live in the $E_\infty^{3,0}, E_\infty^{2,1}, E_\infty^{1,2}$ entries, etc. In particular, $B_\alpha$ and $C_\alpha$ live in the $E_\infty^{2,0}, E_\infty^{3,0}$ entries and are (pullbacks of) the generators of the point group cohomology ring.

  An exception for the elements in the $E_\infty^{0,1}, E_\infty^{0,2}, E_\infty^{0,3}$ entries: instead of labeling them by $A_\beta, B_\gamma, C_\delta$, we label them according to their image in $H^{1,2,3}(T, \mathbb{Z}_2)$ under the restriction map $i^*: H^*(G, \mathbb{Z}_2) \to H^*(T, \mathbb{Z}_2)$ induced by the inclusion $i: T \to G$. For example, if $i^*: B_\gamma \to A_z(A_x + A_y)$, then we label $B_\gamma$ as $B_{z(x+y)}$ instead. Since $E_2^{0,3} = \mathbb{Z}_2$, there can be at most one generator $C$ associated with $E_\infty^{0,3}$, which we label as $C_{xyz}$ following $i^*: C_{xyz} \to A_x A_y A_z$. For degree-1 generators, the labeling may be ambiguous for a small number of space groups and must be treated with care, as discussed in Footnote 12.

- The "Relations" must be factored out and treated as zero in the ring. All relations of the mod-2 cohomology ring of point group $P$—listed in Table II—should survive as (possibly not independent) relations of the mod-2 cohomology ring of any space group $G$ whose associated point group is $P$.

- We have defined the generators of mod-2 cohomology rings such that isomorphic space groups (listed in Eq. (4)) have isomorphic mod-2 cohomology rings.

### 1. Example: No. 227 ($Fd\bar{3}m$)

To illustrate our code and our results, let us consider No. 227 ($Fd\bar{3}m$). This group is generated by three translations $T_{1,2,3}$ as given in Eqs. (92), a two-fold rotation $C_2$, a two-fold rotation $C_2'$, a three-fold rotation $C_3$, a mirror $M$, and an inversion $I$:

$$C_2: (x, y, z) \to (-x + 1/4, -y + 1/4, z), \tag{56a}$$
$$C_2': (x, y, z) \to (-x + 1/4, y, -z + 1/4), \tag{56b}$$
$$C_3: (x, y, z) \to (z, x, y), \tag{56c}$$
$$M: (x, y, z) \to (y, x, z), \tag{56d}$$
$$I: (x, y, z) \to (-x, -y, -z). \tag{56e}$$

An element in No. 227 ($Fd\bar{3}m$) can be written as $T_1^x T_2^y T_3^z C_2^c C_2'^{c'} C_3^{c_3} M^m I^i$ with $x, y, z \in \mathbb{Z}, c, c', m, i \in \{0, 1\}, c_3 \in \{0, 1, 2\}$, hence the cocycles we write down is a function whose arguments are copies of the tuple $(x, y, z, c, c', c_3, m, i)$.

In a GAP interface, we can compute its mod-2 cohomology ring using the command [26]

```
gap> SpaceGroupCohomologyRingGapInterface(227);
```

and the result is

$$\mathbb{Z}_2[A_i, A_m, B_\alpha, B_{xy+xz+yz}, C_\alpha, C_\gamma]/\langle \mathcal{R}_3, \mathcal{R}_4, \mathcal{R}_5, \mathcal{R}_6\rangle \tag{57}$$

where the relations are

$$\mathcal{R}_3: \quad A_i B_\alpha, \tag{58a}$$
$$\mathcal{R}_4: \quad A_i C_\alpha, \quad A_m C_\alpha, \quad A_i C_\gamma, \quad B_\alpha B_{xy+xz+yz} + A_m C_\gamma, \quad B_{xy+xz+yz}(A_i^2 + A_i A_m + B_{xy+xz+yz}), \tag{58b}$$
$$\mathcal{R}_5: \quad B_{xy+xz+yz} C_\alpha, \quad B_{xy+xz+yz} C_\gamma, \tag{58c}$$
$$\mathcal{R}_6: \quad C_\gamma(C_\alpha + C_\gamma). \tag{58d}$$

Here $A_i, A_m, B_\alpha, C_\alpha$ are generators of the point group $O_h$ (see Table II), $B_{xy+xz+yz}$ is a generator coming from $E_\infty^{0,2}$ whose pullback to $P1$ is $A_x A_y + A_x A_z + A_y A_z$, and $C_\gamma$ is a generator coming from $E_\infty^{1,2}$.

### C. Upper bound on the degrees of independent generators and relations

Our code can only deal with finite degrees. Motivated by Theorem 4, we ask: *for 3D space groups, what is the upper bound on the degree at which an independent generator or an independent relation can appear?* We make the following conjecture.

**Conjecture 10.** *(Upper bound of degrees of generators and relations for 3D space groups) For the mod-2 cohomology of 3D space groups, an independent generator can appear at most at degree 6, and an independent relation can appear at most at degree 12.*

For any 3D space group (with the exception of No. 226), our code together with the analysis in Appendix E can at least generate a sub-ring of the full mod-2 cohomology ring excluding potential generators above degree 6. However, whenever such generator of degree $n, n \geq 7$ appears, the $\mathbb{Z}_2$ rank of $H^n(G, \mathbb{Z}_2)$ should be different from the $\mathbb{Z}_2$ rank of the sub-ring at degree $n$. We have explicitly verified that [13] for each space group $G$ (with the exception of No. 226) the $\mathbb{Z}_2$ ranks of $H^n(G, \mathbb{Z}_2)$ derived from the mod-2 cohomology ring presented in Appendix F are exactly the same as the $\mathbb{Z}_2$ ranks directly output by GAP [14] at degree 15 or lower. More specifically, in Table IV we give the generating function (the "Poincaré series") for the ranks of mod-2 cohomology at arbitrary degree $n$ obtained from GAP. We further seek to write down the monomials of the mod-2 cohomology at degree $n$ using generators of the mod-2 cohomology ring we obtain from our code. It turns out that the $\mathbb{Z}_2$ ranks obtained using this method (i.e. the number of monomials) indeed agree with the prediction of the Poincaré series.

Note that once we know the upper bound on the degrees of generators, from the analysis of the standard LHS spectral sequence in Sec. III A, we can directly obtain the upper bound on the degrees of relations, which is simply twice the upper bound on the degrees of generators. This outlines a conceptual way to check the Conjecture 10. Motivated by explicit LHS spectral sequence calculations we have performed (see Sec. III A for an example), we make the following conjecture which can be used to estimate the maximal degree of cohomology ring generators.

**Conjecture 11.** *(Module structure of the $q = 1, 2$ rows of the LHS spectral sequence) Given any 3D space group $G$, the module generator of the $q = 1$ row and the $q = 2$ row of the standard LHS spectral sequence at $E_2$ page can only appear at the $p = 0$ column or $p = 1$ column.*

This conjecture can be checked by a direct calculation for all 230 space groups. In fact, one can restrict the calculation to a much smaller set. First, note that the action of the point group $P$ on translations $T$ is the same for all space groups in the same arithmetic crystal class listed in Table II. As a result one only needs to perform the calculation once for each of the 73 arithmetic crystal classes. Second, since inversion $I$ will always change a translation generator to its inverse and hence will not act on $H^{1,2}(T, \mathbb{Z}_2) \cong \mathbb{Z}_2^3$, adding inversion $I$ will not give rise to new module generators. Therefore, one can ignore all arithmetic crystal classes involving inversion $I$. Furthermore, a single calculation works for different point groups that are isomorphic to the same abstract groups, as long as the action of the point groups on $H^q(T, \mathbb{Z}_2)$ is identical. This also significantly reduces the number of cases we need to check.

### D. Some interesting features

In this subsection, we collect several intriguing aspects of the cohomology of space groups.

1. Almost all groups have generators of mod-2 cohomology ring in degrees 3 or lower, but there are several groups that contain generators of higher degrees. Specifically, the following 10 space groups have degree-4 generators:

$$108\ (I4cm), \quad 109\ (I4_1md), \quad 120\ (I\bar{4}c2), \quad 130\ (P4/ncc), \quad 136\ (P4_2/mnm), \tag{59}$$
$$140\ (I4/mcm), \quad 142\ (I4_1/acd), \quad 197\ (I23), \quad 204\ (Im\bar{3}), \quad 230\ (Ia\bar{3}d).$$

   No group has degree-5 generators. The following 3 space groups have degree-6 generators:

$$219\ (F\bar{4}3c), \quad 226\ (Fm\bar{3}c), \quad 228\ (Fd\bar{3}c). \tag{60}$$

2. There are ten non-isomorphic Bieberbach groups, i.e., torsion-free crystallographic groups, in 3D,

$$1\ (P1), \quad 4\ (P2_1), \quad 7\ (Pc), \quad 9\ (Cc), \quad 19\ (P2_12_12_1), \quad 29\ (Pca2_1), \quad 33\ (Pna2_1), \tag{61}$$
$$76/78\ (P4_1/P4_3), \quad 144/145\ (P3_1/P3_2), \quad 169/170\ (P6_1/P6_5).$$

   These groups are labeled with a "♭" in the third column of Table IV. For all these Bieberbach groups, we have $H^3(G, \mathbb{Z}_2) = \mathbb{Z}_2$ and $H^n(G, \mathbb{Z}_2) = 0, n \geq 4$, consistent with Theorem 7. According to Sec. V, the (unique) nontrivial element in $H^3(G, \mathbb{Z}_2)$ corresponds precisely to the fundamental domain, which is the unique IWP for these Bieberbach groups.

3. In 3D, the point groups satisfying the condition of admitting a periodic free $\mathbb{Z}P$ resolution of $\mathbb{Z}$ are

$$C_i, C_2, C_s \cong \mathbb{Z}_2, \quad C_4, S_4 \cong \mathbb{Z}_4, \quad C_3 \cong \mathbb{Z}_3, \quad D_3, C_{3v} \cong Dih_3, \quad S_6, C_6, C_{3h} \cong \mathbb{Z}_6. \tag{62}$$

---

[13] The code used for verification can be found in the Mathematica file `Space_Group_Cohomology_Data.nb` in our Github repository [26].

[14] The GAP command `ResolutionSpaceGroup` can be used to directly obtain the rank of the mod-2 cohomology for space groups (but not the mod-2 cohomology ring structure, as this command does not yet implement contracting homotopy).

The period is one for $\mathbb{Z}_2$, two for $\mathbb{Z}_4, \mathbb{Z}_3, \mathbb{Z}_6$, and four for $Dih_3$, and these groups have the same mod-2 cohomology in all positive degrees [40, 47]. According to Corollary 8, we see that indeed all the associated space groups have identical cohomology in degrees greater than 3, which can be straightforwardly checked in Table IV.

In 3D, 109 out of the 230 space groups admit a periodic resolution at degrees larger than 3. Their $\mathbb{Z}_2$ ranks of mod-2 cohomology all have period one, i.e., $H^n(G, \mathbb{Z}_2) \cong H^{n+1}(G, \mathbb{Z}_2)$, for $n \geq 4$ (see the last column of Table IV).

4. All point groups associated with space groups No. 143–230 contain three-fold rotations. For any $G$ in No. 143–230, we can choose a space group $G_{\mathcal{Z}}$ as a subgroup of $G$ of index 3. $G_{\mathcal{Z}}$ can be thought of as $G$ discarding three-fold rotations, and is one of the space groups No. 1–142.

We claim that

**Theorem 12.** *The embedding* $i\colon G_{\mathcal{Z}} \to G$ *induces an injective ring homomorphism* $i^*\colon H^*(G, \mathbb{Z}_2) \hookrightarrow H^*(G_{\mathcal{Z}}, \mathbb{Z}_2)$.

*Proof.* The proof directly follows the proof for finite groups in Ref. [48]. Consider the transfer (i.e. corestriction) map $\mathrm{Tr}\colon H^*(G_{\mathcal{Z}}, \mathbb{Z}_2) \to H^*(G, \mathbb{Z}_2)$. The composition of $i^*$ with the transfer map is simply multiplication by 3. Hence, if for some $u \in H^*(G, \mathbb{Z}_2)$ we have $i^*(u) = 0$, composing with the transfer map we have $3u = 0$ and hence $u = 0$, proving the injectivity of $i^*$. $\qquad\square$

In addition, if $G_{\mathcal{Z}} \lhd G$ is a normal subgroup of $G$, elements in $H^*(G, \mathbb{Z}_2)$ are precisely those elements in $H^*(G_{\mathcal{Z}}, \mathbb{Z}_2)$ which are invariant under the induced action of the three-fold rotation $C_3$. This fact will be important when we deal with several groups whose relations cannot be straightforwardly obtained by our codes in Appendix E.

## V. Deriving Lieb–Schultz–Mattis (LSM) constraints

Equipped with knowledge of the mod-2 cohomology of crystallographic groups, we are now well-positioned to investigate various LSM constraints in 3D systems. As we will show below, for every IWP of the space group $G$, there is an element in the third group cohomology $H^3(G, \mathbb{Z}_2)$ that can be uniquely assigned to this IWP. This correspondence between IWPs and cohomology elements underlies the physics of LSM constraints when the coefficient, $\mathbb{Z}_2$, of the cohomology $H^3(G, \mathbb{Z}_2)$ admits the meaning of a classification of *anomalous textures* [15] for the on-site degrees of freedom at this IWP: when the system has an additional internal symmetry, and the on-site degrees of freedom form a projective representation of this internal symmetry correspond to the nontrivial element of the $\mathbb{Z}_2$ (hence an *anomalous texture*), an *LSM anomaly* is triggered, forbidding the physical system to have a unique, symmetric, gapped ground state.

Here, following the tradition of LSM theorems, we consider on-site degrees of freedom with a single $\mathbb{Z}_2$ classification under the action of internal symmetry. The most familiar example is spins with SO(3) rotation symmetry, where half-integer (resp. integer) spins correspond to a projective (resp. faithful) representation of SO(3) labeled by the nontrivial (resp. trivial) element of $H^2(\mathrm{SO}(3), U(1)) \cong \mathbb{Z}_2$.

Other internal symmetry— such as the $\mathbb{Z}_2 \times \mathbb{Z}_2$ subgroup of SO(3) generated by two $\pi$ rotations along two perpendicular axes, or an anti-unitary time-reversal symmetry $\mathbb{Z}_2^{\mathcal{T}}$—can also lead to a $\mathbb{Z}_2$ classification for the onsite degree of freedom, the nontrivial class of which contains LSM constraints [49]. In this sense, the analysis of LSM constraints pertains to a broad class of models—including any exchange Hamiltonian written in terms of spin-1/2 operators in absence of an external field and with or without spin–orbit coupling (XXZ, Dzyaloshinskii–Moriya, dipole-dipole, dipole-octopole etc.). For all these cases, the projective representation of the internal group $G_{\mathrm{int}}$ triggers an LSM anomaly as a descendent of certain element of $H^5(G \times G_{\mathrm{int}}, U(1)^{\mathrm{or}})$, which classifies all the ('t Hooft) anomalies associated with the global symmetry $G \times G_{\mathrm{int}}$.

For clarity, we will primarily assume the internal symmetry to be SO(3) in the rest of discussions; the results however apply to all the internal symmetries mentioned above.

### A. LSM(OH) theorem for translation symmetries

The Lieb–Schultz–Mattis theorem in 1D, along with its later generalization to higher dimensions by Oshikawa and Hastings, provides important constraints on the ground state of a many-body lattice system's Hamiltonian based solely on basic symmetry properties, without relying on other specific details of the Hamiltonian. For simplicity, we will focus on the case where the system exhibits on-site SO(3) symmetry without spin–orbit coupling. A key feature of SO(3) symmetry is that its projective representations are classified by $\mathbb{Z}_2$, corresponding to half-integer or integer spins in physical terms. This discussion can be readily extended to cases where the projective representation of the internal symmetry is classified by $(\mathbb{Z}_2)^k$. We will briefly comment on situations where the projective representation is $\mathbb{Z}_3$-classified, with a more comprehensive treatment left for future work.

————

[15] We refrain from providing a precise definition of *anomalous textures* and instead refer the reader to Ref. [35] for a detailed exposition.

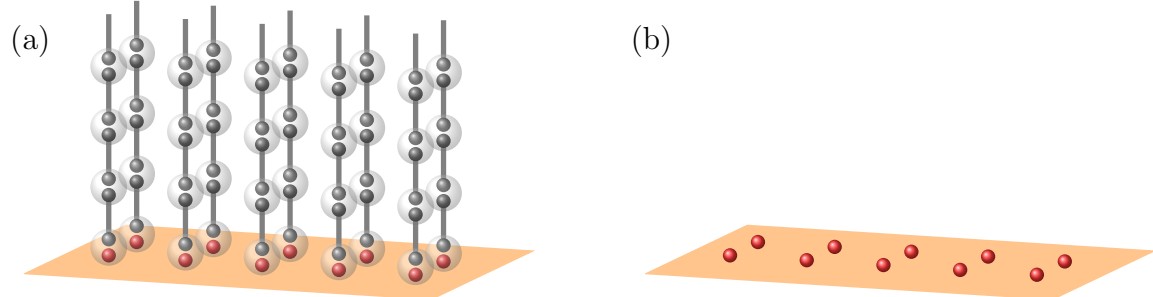

FIG. 3. Bulk-edge correspondence for $k$-dimensional spin-1/2 lattice (the figure illustrates $k = 2$). The on-site spin-1/2 lives on the boundary of a fictitious $(k+1)$-dimensional bulk. (a): the $(k+1)$-dimensional bulk. The bulk SPT consists of $S = 1$ Haldane chains forming an $k$-dimensional array. The boundary of each Haldane chain has an $S = 1/2$ edge mode (red dots). (b): the $k$-dimensional boundary on which the on-site $S = 1/2$ degrees of freedom live.

**Statement 13.** *(Lieb–Schultz–Mattis–Oshikawa–Hastings [13, 50, 51]) In a crystal with translation symmetry and on-site* SO(3) *symmetry, if there are odd numbers of spin-1/2 degrees of freedom per unit cell, then there cannot be a unique, symmetric, gapped ground state.*

This constraint on the ground state wave function resembles that of the 't Hooft anomaly on the ground state wave function or the low-energy subspace of a quantum field theory (QFT). This connection was later explored in [14, 52, 53], identifying LSMOH constraints as a mixed anomaly between (lattice) translation and internal SO(3) symmetries.

According to the Crystalline Equivalence Principle [20, 35], the 't Hooft anomaly is characterized by an element in

$$H^{k+2}(\mathbb{Z}^k \times \mathrm{SO}(3), \mathrm{U}(1)) \cong \mathbb{Z}_2, \tag{63}$$

where the nontrivial element can be written as

$$\exp(i\pi A_x A_y \cdots \cup w_2^{\mathrm{SO}(3)}). \tag{64}$$

Here $\exp(i\pi \cdots)$ maps an element $\lambda \in H^{k+2}(G, \mathbb{Z}_2)$ to an element $\exp(i\pi\lambda) \in H^{k+2}(G, \mathrm{U}(1))$. $w_2^{\mathrm{SO}(3)}$ is the generator of $H^2(\mathrm{SO}(3), \mathbb{Z}_2) \cong \mathbb{Z}_2$ and $\exp(i\pi w_2^{\mathrm{SO}(3)}) \in H^2(\mathrm{SO}(3), \mathrm{U}(1)) \cong \mathbb{Z}_2$ is the cohomology element corresponding to the projective representation of SO(3), i.e. spin-1/2. $A_x A_y \cdots$ is the generator of $H^k(\mathbb{Z}^k, \mathbb{Z}_2) \cong \mathbb{Z}_2$ and $A_x, A_y, \cdots$ are the generators of each individual $H^1(\mathbb{Z}, \mathbb{Z}_2)$, which can be roughly interpreted as the gauge fields corresponding to translations in each direction.

This cohomology element can be understood from the perspective of anomaly-inflow [53]. From this perspective, consider placing the $k$-dimensional spin-1/2 system on the boundary of an $(k + 1)$-dimensional crystalline SPT, constructed by stacking SO(3) Haldane chains, i.e., $(1 + 1)$-dimensional SPTs protected by SO(3) symmetry, while respecting the translation symmetries along the original $n$ directions, as illustrated in Figure 3. The crystalline SPT is clearly characterized by the cohomology element in Eq. (64), which can be thought of as the partition function of the crystalline SPT (or its TQFT), if we integrate it over the $(k+2)$-dimensional spacetime $\mathcal{M}_{k+2}$ and write it as $\exp(i\pi \int_{m_{k+2}} A_x A_y \cdots \cup w_2^{\mathrm{SO}(3)})$.

Now we slice a boundary for this crystalline SPT perpendicular to the extra dimension. Because the boundary of each SO(3) Haldane chain has a spin-1/2 edge mode, the boundary is exactly characterized by a lattice of spin-1/2 degrees of freedom with one spin-1/2 per unit cell. Hence, from the perspective of anomaly-inflow, we see that the original system carries an anomaly characterized by Eq. (64).

We mention that the cohomology element Eq. (64) can be derived from the *anomalous texture* of the lattice system [35], using the isomorphism between equivariant homology and group cohomology. In 1D, this cohomology element can also be understood through gauging [54, 55] or the *anomaly index* of symmetry actions [56].

### B. LSM constraints for general crystallographic groups and IWP

We now explore the general conditions for a generic crystallographic group to give rise to LSM constraints, in particular in 3D.

Recall from Section II B that, performing lattice homotopy, which is a smooth deformation of the lattice structure that should not change the anomaly of the system, puts all physical degrees of freedom to the Irreducible Wyckoff Positions (IWP) of the crystallographic group $G$. It is then conjectured in Ref. [34] that

**Conjecture 14.** *(General LSM constraints [34]) A symmetric short range entangled (sym-SRE) phase is possible only when the lattice is smoothly deformable to a trivial lattice.*

In a "modern" point of view, the LSM theorem is a statement about *anomalous texture cancellation condition* [35]: the anomaly carried by the microscopic degree of freedom (the "anomalous texture") must be cancelled by the anomaly of the macroscopic phase of the matter (or more precisely, the "defect network") to make the system anomaly-free.

The traditional LSM theorem is the statement that: The ground state is noninvertible (i.e. has symmetry enriched topological order) if and only if the equivariant pushforward map

$$H^2(G, \Lambda = Z_0(\mathbb{R}^d, \mathrm{U}(1))) \hookrightarrow \mathcal{H}^G_{-2}(X = \mathbb{R}^d, \mathrm{U}(1)) \cong H^{d+2}(G, \mathrm{U}(1)), [\omega] \mapsto [\mu] \tag{65}$$

has nontrivial image, i.e. $[\mu]$ is a genuine cocycle and not a coboundary. In words, $[\omega] \mapsto [0]$ means that the (anomaly of the) anomalous texture can be canceled by (the anomaly of some) invertible-substrate defect network, while $[\omega] \mapsto [\mu] \neq [0]$ means that the (anomaly of the) anomalous texture cannot be canceled by (the anomaly of any) invertible-substrate defect network, and hence the ground state has an anomaly (and hence is invertible/has topological order).

In principle, it can happen that the map $[\omega] \mapsto [0] \in H^G_{-2}(X, \mathrm{U}(1))$ is trivial (i.e. no traditional LSM), but the map $[\omega] \mapsto [\mu_1] \neq [0] \in H^G_{-2}(X_1, \mathrm{U}(1))$ is nontrivial (in the sense of lattice homotopy). The main result of Ref. [35] is that, by explicit checking, the authors find that for specific forms of $\mathcal{G}$ in the form of $\mathcal{G} = G \times \mathrm{SO}(3)$ (and two other cases),

$$[\omega] \mapsto [\mu] = [0] \Leftrightarrow [\omega] \mapsto [\mu_1] = [0], \tag{66}$$

hence lattice homotopy can serve as the equivalent criterion of LSM criterion, i.e. having a nontrivial lattice homotopy is equivalent to having a noninvertible ground state/topological order.

From the form of the equivariant pushforward map, to generalize to crystalline symmetries beyond pure translations, we postulate that given any lattice symmetry $G$, we can associate the lattice structure by an element $\lambda \in H^n(G, \mathbb{Z}_2)$ corresponding to the IWPs where projective representations live. Then the corresponding element in group cohomology is given by

$$\exp\left(i\pi\lambda \cup w_2^{\mathrm{SO}(3)}\right). \tag{67}$$

For general internal symmetry groups $G_{\mathrm{int}}$, we just need to change $w_2^{\mathrm{SO}(3)}$ to the element in $H^2(G_{\mathrm{int}}, \mathrm{U}(1))$ that characterizes the projective representation of the on-site symmetry. The main part of latter discussion is to identify this $\lambda$ as an element in the cohomology ring.

## C.  LSM constraints in 2D

Specializing to 2D, the cohomology element in $H^2(G, \mathbb{Z}_2)$ corresponding to the IWPs is worked out in [15] in an explicit way. In this subsection, we review these results with an eye toward generalizing them to 3D scenarios.

Given a wallpaper group symmetry, its IWP can be decomposed into the following three classes based on its little group, i.e., subgroup of the wallpaper group that leaves individual points of the IWP invariant:

- A fundamental domain that tiles the 2D space under the actions of translation and glide symmetries, with the corresponding little group $C_1$ (trivial),

- A translation unit cell along a mirror axis, with the corresponding little group $D_1$,

- An $n$-fold rotation center, with the corresponding little group either $C_n$ or $D_n$.

Here $C_n$ and $D_n$ are groups of $n$-fold rotation, and $n$-fold rotation together with reflection. In particular, $D_1$ is reflection only.

We point out that we can simply view these three classes as three no-go theorems for symmetric short range entangled (SRE) states. Namely, from lattice homotopy, if there is a nontrivial $\mathbb{Z}_2$-classified projective representation in each class, then except the case where $n = 3$ in the third item, they all give rise to nontrivial LSM constraints and a symmetric-SRE is forbidden [34]. In particular, if there is a nontrivial $\mathbb{Z}_2$-classified projective representation in any IWP (possibly after the operation of lattice homotopy), a nontrivial LSM constraint is triggered and a symmetric-SRE is forbidden.

**Statement 15.** *(LSM constraints in 2D [34]) A 2D LSM no go theorem is triggered when and only when there are odd numbers of spin-1/2's in any of the three categories of IWPs listed above.*

To identify an element in $H^2(G, \mathbb{Z}_2)$ corresponding to these IWPs, first we mention that the elements in $H^2(G, \mathbb{Z}_2)$ can be identified by its dual *homology element*, or chain representative, in $H_2(G, \mathbb{Z}_2)$. The action of some cohomology element on a specific homology element will give a function of the cohomology element, which is called *topological invariant* in [15].

Specializing to $G$ being a wallpaper group, we need to identify the *minimal* set of group elements, on which the topological invariants completely determine the IWP. Given the three types of IWPs above, we have the following three possibilities:

1. Given two commuting elements $g_1, g_2 \in G$ such that $g_1 g_2 = g_2 g_1$, we can define the following topological invariant,

$$\phi_2[g_1, g_2] := \lambda(g_1, g_2) + \lambda(g_2, g_1). \tag{68}$$

- When $g_1$ and $g_2$ are two translations $T_{1,2}$ in two directions, this topological invariant $\phi_2[T_1, T_2] = \lambda(T_1, T_2) + \lambda(T_2, T_1)$ corresponds to the first class of IWPs.

- When $g_2$ is a reflection $M$ and $g_1$ is a translation $T$ along the reflection axis, this topological invariant $\phi_2[T, M] = \lambda(T, M) + \lambda(M, T)$ corresponds to the second class of IWPs.

2. Given an order-2 element $g \in G$ such that $g^2 = 1$, we can define the following topological invariant

$$\phi_1[g] := \lambda(g, g) + \lambda(1, 1). \tag{69}$$

- When $g$ is a $C_2$ rotation, this topological invariant corresponds to the third class of IWPs

In addition, the Bieberbach group $pg$ is special, whose unique IWP is the fundamental domain generated by a translation $T$ and a glide symmetry $G$. Its topological invariant, or simply the chain representative for the nontrivial element in $H_2(pg, \mathbb{Z}_2)$ has a slightly complicated form:

$$\widetilde{\phi}_2[T, G] = \lambda(TG, TG^{-1}) + \lambda(T, G) + \lambda(T, G^{-1}) + \lambda(G, G^{-1}). \tag{70}$$

Finally, in order to identify an element in $H^2(G, \mathbb{Z}_2)$, we need a complete list of chain representatives or topological invariants of $H_2(G, \mathbb{Z}_2)$. It turns out that for 2D wallpaper groups, there is only one more possibility:

$$\phi_1[M] = \lambda(M, M) + \lambda(1, 1), \tag{71}$$

where $M$ is the mirror reflection. We can immediately see that this is a topological invariant involving $M$ only, yet the reflection $M$ cannot specify an IWP. We will call this the non-LSM invariant. It is a highly nontrivial check that, for all wallpaper groups $G$, the three topological invariants corresponding to IWPs together with the one extra non-LSM invariant completely spans all elements in $H_2(G, \mathbb{Z}_2)$, hence completely specify elements in $H^2(G, \mathbb{Z}_2)$.

Hence, to associate every IWP of wallpaper groups with an element in $H^2(G, \mathbb{Z}_2)$, we need to (1) write down the complete list of topological invariants for $G$, including one for every IWP and all the non-LSM invariants, the number should be equal to the rank of $H^2(G, \mathbb{Z}_2)$; (2) find out the element in $H^2(G, \mathbb{Z}_2)$ whose value on the associated topological invariant is one, while zero on all the other topological invariants. This procedure is explicitly carried out in Ref. [15] for all wallpaper groups.

In 2D the identification of an IWP and an element in $H^2(G, \mathbb{Z}_2)$ can be made more rigorous by the argument of "inserting a flux" or either SO(3) rotation symmetry [15] or wallpaper group symmetry $G$ [34]. Specifically, we can consider coupling the system to a probe gauge field of the SO(3) symmetry and examine the flux, or monopole, of this SO(3) gauge field. Because the wave function of the system acquires a $-1$ phase factor when an SO(3) monopole circles around a Haldane chain, if any of the 3 basic no-go theorems is triggered, the SO(3) monopole will carry a specific projective representation of $G$. Specifically, for the three types of IWPs, $T_1 T_2 T_1^{-1} T_2^{-1}$, $TMT^{-1}M^{-1}$ and $C_2^2$ acting on SO(3) will generate an extra $-1$ sign. Hence, we can view $\lambda \in H^2(G, \mathbb{Z}_2)$ as the projective representation, or the symmetry fractionalization pattern, of the SO(3) monopole, which completely encodes the LSM constraint.

Finally, we have the following observation about the subgroup of $H^2(G, \mathbb{Z}_2)$ containing cohomology elements associated with LSM constraints,

**Statement 16.** *(LSM anomaly in 2D) 2D LSM constraints are associated with the subgroup* $\ker \beta_2$ *of* $H^2(G, \mathbb{Z}_2)$*, that is, the kernel of the Bockstein homomorphism:*

$$\beta_2 \colon H^2(G, \mathbb{Z}_2) \to H^3(G, \mathbb{Z}^{or}). \tag{72}$$

For 2D wallpaper groups $G$, this condition is isomorphic to the condition that they vanish under the action of $\mathcal{S}\mathcal{Q}^1 := Sq^1 + A_m\cup$, defined in Eq. (38). Ref. [15] gives an argument by restricting to the U(1) subgroup of SO(3).

### D. LSM constraints in 3D

In this subsection, we go to 3D and outline the algorithm to identify the cohomology element in $H^3(G, \mathbb{Z}_2)$ corresponding to the IWPs of every 3D space groups. The full result for all 3D space groups is collected in the table in Appendix F.

Similar to the 2D scenarios, in 3D lattices the IWPs can be classified into the following five categories:

- A fundamental domain that tiles the 3D space under the actions of translation, glide and screw symmetries,

- A fundamental domain that tiles a 2D reflection plane under the actions of translation, glide and screw symmetries,

- A translation unit cell along a $C_2$ rotation axis,

- An intersection point of two perpendicular $C_2$ rotation axes,

- An inversion center.

We see that they roughly correspond to codimension 0, 1, 2, and 3 (for the last two items) regions in the Euclidean space. Here we conjecture that they *all* give rise to nontrivial LSM constraints.

**Statement 17.** *(LSM constraints in 3D.) A 3D LSM no go theorem is triggered when and only when there are odd numbers of spin-1/2's on any of the five categories of IWPs listed above.*

Now, in 3D, our task is to associate an element in $H^3(G, \mathbb{Z}_2)$ to every IWP for all 3D space groups. To achieve this task, we define the following three types of topological invariants in parallel to our considerations in 2D:

1. Given three mutually commuting elements $g_1, g_2, g_3 \in G$, we define the following topological invariant

$$\varphi_3[g_1, g_2, g_3] := \lambda(g_1, g_2, g_3) + \lambda(g_1, g_3, g_2) + \lambda(g_2, g_1, g_3) + \lambda(g_2, g_3, g_1) + \lambda(g_3, g_1, g_2) + \lambda(g_3, g_2, g_1). \tag{73}$$

   - When $g_1$, $g_2$ and $g_3$ are translations in three directions, this topological invariant corresponds to the first class of IWPs.
   - When $g_3$ is reflection and $g_1$ and $g_2$ are translations along two directions in the reflection plane, this topological invariant corresponds to the second class of IWPs.

2. Given two group commuting elements $g_1, g_2 \in G$ such that $g_2^2 = 1$ and $g_1 g_2 = g_2 g_1$, we define the following topological invariant,

$$\varphi_2[g_1, g_2] := \lambda(g_1, g_2, g_2) + \lambda(g_2, g_1, g_2) + \lambda(g_2, g_2, g_1) + \lambda(g_1, 1, g_1). \tag{74}$$

   - When $g_2$ is a $C_2$ rotation and $g_1$ is translation/glide/screw along the rotation axis, this topological invariant corresponds to the third class of IWPs.
   - When $g_1$ and $g_2$ are two $C_2$ rotations along two perpendicular axes, this topological invariant corresponds to the fourth class of IWPs.

3. Given an order-2 element $g \in G$ such that $g^2 = 1$, we define the following topological invariant

$$\varphi_1[g] := \lambda(g, g, g) + \lambda(g, 1, g). \tag{75}$$

   - When $g$ is inversion, this topological invariant corresponds to the fifth class of IWPs.

In addition, we should consider Bieberbach groups separately. The topological invariant for each Bieberbach group is special, and it is listed in Appendix F under the table for each Bieberbach group separately.

For topological invariants not associated with IWP, i.e., the non-LSM invariants, we can restrict to the following four possibilities:

- $\varphi_1[C_2]$, where $C_2$ is some two-fold rotation,

- $\varphi_1[M]$, where $M$ is some reflection,

- $\varphi_2[X, M]$, where $X$ is any generator that commutes with reflection $M$,

- $\varphi_2[C_4, C_4^2]$, where $C_4$ is some four-fold rotation.

We immediately see that for these non-LSM invariants, the group elements involved cannot specify an IWP. It turns out that the listed topological invariants combined, including LSM and non-LSM topological invariants, are enough to determine $H^3(G, \mathbb{Z}_2)$ completely, for every 3D space group $G$, which is a highly nontrivial consistency check.

Hence, to associate every IWP of 3D space groups $G$ with an element in $H^3(G, \mathbb{Z}_2)$, we have the algorithm (1) write down the complete list of topological invariants for $G$, including one for every IWP and all the non-LSM invariant, the number should be equal to the rank of $H^3(G, \mathbb{Z}_2)$; (2) find out the element in $H^3(G, \mathbb{Z}_2)$ whose value on the associated topological invariant is one, while nonzero on all the other topological invariants. We implement this algorithm in our code and this indeed gives the cohomology element associated with every IWP, and the results are collected in each table in Appendix F. The tables contain basic information about all IWPs, including its name in ITC, its associated little group, and its coordinates. In the last two entries, we write down the topological invariant for each IWP, directly from the little group, as well as the cohomology class associated with the IWP.

As stated in the main text, given the internal symmetry to be e.g. SO(3), the value of the topological invariant for a given cohomology element in $H^3(G, \mathbb{Z}_2)$ can detect whether there is a spin-1/2 degree of freedom at the IWP. If we also insert a flux for SO(3) rotation symmetry in 3D, now the monopole should not be a point operator in 2D space, but a line in 3D. We can also analyze the symmetry fractionalization of this monopole, whose symmetry fractionalization pattern should be classified by an element in $H^3(G, \mathbb{Z}_2)$. This element is exactly the element associated with the IWP.

Finally, we also mention the following observation,

**Statement 18.** *(LSM Anomaly in 3D) 3D LSM constraints are associated with the subgroup* $\ker \beta_2 \cap \ker \mathcal{SQ}^2$ *of* $H^3(G, \mathbb{Z}_2)$, *that is, the intersection of kernels of the following two maps:*

$$\beta_3 \colon H^3(G, \mathbb{Z}_2) \to H^4(G, \mathbb{Z}^{or}), \tag{76}$$

*and*

$$\mathcal{SQ}^2 \colon H^3(G, \mathbb{Z}_2) \to H^5(G, \mathbb{Z}_2). \tag{77}$$

*Here,* $\beta_3$ *is the Bockstein homomorphism associated with the short exact sequence* $\mathbb{Z}^{or} \to \mathbb{Z}^{or} \to \mathbb{Z}_2$ *with* $\mathbb{Z}^{or}$ *the orientation-reversing* $\mathbb{Z}$ *module.* $\mathcal{SQ}^2$ *is defined as* $Sq^2 + w_1 \cup Sq^1 + (w_2 + w_1^2) \cup$, *where* $w_1 \in H^1(G, \mathbb{Z}_2)$ *and* $w_2 \in H^2(G, \mathbb{Z}_2)$ *are the pullback of Stiefel-Whitney classes for the natural 3-dimensional representation of* $P$ *acting on the 3D Euclidean space (see Table II).*

Interestingly, $\mathcal{SQ}^1$ (defined in Eq. (38)) and $\mathcal{SQ}^2$ are Steenrod operations on Thom space of $BG$ equipped with the 3-dimensional vector bundle explained in the statement, that is, the pullback of the natural 3-dimensional representation of $P$ acting on 3D Euclidean space [16]. Notably, $\mathcal{SQ}^2$ appears in the differential of the Atiyah-Hirzebruch spectral sequence, which classifies fermionic crystalline SPTs with spinless fermions, according to the fermionic Crystalline Equivalence Principle [23]. It is intriguing to see how this fact may influence efforts to generalize LSM theorems to crystalline systems featuring on-site Majorana fermions [57–59].

### E. Examples

Now, through a few examples, we illustrate how to extract the data of LSM constraints from the tables we provide in Appendix F. We also highlight certain interesting features of the groups considered in these examples.

#### 1. No. 16: P222

No. 16 ($P222$) is generated by three translations and two two-fold rotations along two perpendicular axes. According to the IWP Table of No. 16, there are in total eight IWPs, corresponding to one site, three edge centers, three face centers, and one body center in a type-$P$ Bravais lattice, which are precisely eight intersection points of two two-fold rotations in a unit cell.

From the Table, we can immediately write down the cohomology element in $H^3(G, \mathbb{Z}_2)$ for $2^8 = 256$ different kinds of LSM constraints for No. 16. Note that according to Ref. [34], we can obtain 255 LSM constraints from considering lattice homotopy, yet there is one LSM constraint that cannot be derived this way, and it corresponds to putting spin-1/2's on all IWPs. We believe that the last one still corresponds to LSM constraints, and its corresponding cohomology element can be obtained by summing over the last column in the IWP Table of No. 16, which is $A_c A_{c'}(A_c + A_{c'})$.

#### 2. No. 19: P2₁2₁2₁

No. 19 ($P2_12_12_1$) is generated by three translations and two two-fold screws. This group is torsion free, i.e., there is no group element other than identity such that some power of the group element is identity, hence it is a Bieberbach group in 3D. And we have observed that it satisfies $H^3(G, \mathbb{Z}_2) \cong \mathbb{Z}_2$ and $H^k(G, \mathbb{Z}_2)$ are all trivial for $k \geq 4$.

For these Bieberbach groups, the topological invariant, or simply the chain representative for $H_3(G, \mathbb{Z}_2) \cong \mathbb{Z}_2$, is usually very complicated. This is a special feature of these Bieberbach groups. We provide one such candidate in Eq. (195).

From the coordinate entry of the (unique) IWP in the IWP Table of No. 19, we see that every unit cell contains four regions related to each other by screw operations. Hence, every unit cell contains four fundamental domains. Let us restrict to the subgroup of No. 19, say just No. 1 ($P1$). Because each unit cell contains four fundamental domains, the IWP for No. 19 no longer remains to be the IWP for No. 1, thus the restriction of the cohomology element $A_c B_{\beta 1}$ is also zero.

#### 3. No. 143: P3

No. 143 ($P3$) is generated by three translations and a three-fold rotation. We see that there are three IWPs for this group, yet these three IWPs correspond to the same IWP once we remove the three-fold rotation. Hence, when considering

---

[16] Given Eq. (38) where $\beta_3$ appears, it might be visually appealing to change the criterion from $\ker \beta_2 \cap \ker \mathcal{SQ}^2$ to $\ker \mathcal{SQ}^1 \cap \ker \mathcal{SQ}^2$. But the two sets are not exactly the same. This is different from the situation in 2D where, for all wallpaper groups, $\ker \beta_2$ is isomorphic to $\ker \mathcal{SQ}^1$. For example, in space group No. 75 ($P4$), there exists an element $B_\alpha A_{\mathsf{q}} \in H^3(P4, \mathbb{Z}_2)$ that does not correspond to any LSM constraint, yet it is zero under the actions of both $\mathcal{SQ}^1$ and $\mathcal{SQ}^2$. Elements $\lambda \in H^3(G, \mathbb{Z}_2)$ which are zero under $\mathcal{SQ}^1$ and $\mathcal{SQ}^2$ but do not correspond to LSM constraints will yield a value of 1 for the fourth non-LSM topological invariant, i.e., $\varphi_2[C_4, C_4^2]$ for some $C_4$ rotation.

mod-2 cohomology, or when the on-site degree of freedom is $\mathbb{Z}_2$ classified, we need to treat these three IWPs on equal footing. In particular, they correspond to the same cohomology element in $H^3(P3, \mathbb{Z}_2)$.

Still, when the on-site degrees of freedom is $\mathbb{Z}_3$-classified, we will see that the three IWPs correspond to three different cohomology elements, in fact three generators, in $H^3(P3, \mathbb{Z}_3) \cong (\mathbb{Z}_3)^6$. These three elements generate the subgroup $H^3(P3, \mathbb{Z}^{\mathrm{or}}) \otimes \mathbb{Z}_3 \cong (\mathbb{Z}_3)^3$.

### 4. No. 147: $P\bar{3}$

No. 147 ($P\bar{3}$) is generated by three translations, a three-fold rotation and an inversion. There are five IWPs in total. If we forget about the three-fold rotation and restrict to the subgroup $P\bar{1} \subset P\bar{3}$, four of these IWPs still survive as nontrivial IWPs. However, there is one, with coordinate $(1/3, 2/3, z), (2/3, 1/3, -z)$, which does not survive as a nontrivial IWP, because it does not correspond to any inversion center of $P\bar{1}$. Hence, this IWP will not contribute to LSM constraints (for on-site $\mathbb{Z}_2$ degrees of freedom), and corresponds to no element in $H^3(P\bar{3}, \mathbb{Z}_2)$.

When the on-site degree of freedom is $\mathbb{Z}_3$-classified, we see that this IWP corresponds to a nontrivial cohomology element in $H^3(P\bar{3}, \mathbb{Z}_3) \cong (\mathbb{Z}_3)^3$. In particular, this element is one of the two generators for $H^3(P\bar{3}, \mathbb{Z}^{\mathrm{or}}) \otimes \mathbb{Z}_3 \cong (\mathbb{Z}_3)^2$.

## VI.   Anomaly matching and relationship to PSG on the pyrochlore lattice

In this section, to establish a connection with physics of lattice systems, we focus on space group No. 227 ($Fd\bar{3}m$), which contains the crystalline symmetry of both pyrochlore and diamond lattices. To illustrate the application of our results of the LSM anomaly, we present an example that shows how we can use the expression of LSM anomaly to give concrete prediction of the emergent infrared (IR) theory through anomaly matching. Specifically, motivated by Refs. [27], we consider U(1) quantum spin liquids (QSL) on the pyrochlore lattice.

### A.   Review of 3D Quantum Spin Liquid

Let us begin with a brief overview of the physics underlying U(1) quantum spin liquids (QSLs) in three dimensions, in particular the emergent IR field theory and its associated anomaly. Quantum spin liquids are zero-temperature phases of magnets in which the fluctuating quantum spins avoid magnetic long-range order and stay in liquid-like states [60]. These states are fundamentally characterized by intrinsic long-range entanglement: the non-local excitations interact with each other through an emergent gauge field, making gauge theories a natural framework for describing QSLs. Quite often, the elementary excitations in a QSL exhibits the phenomenon of *symmetry fractionalization*—they carry fractional quantum numbers of the full symmetry $\mathcal{G}$ which label the projective representations of $\mathcal{G}$.

Extensive research has been conducted on 2D QSLs, partly driven by the LSM anomaly and anomaly matching [14–17, 61–64]. Notably, given the lattice structure we have been considering, the LSM anomaly should be equal to the anomaly of the emergent IR theory. From this simple idea, Refs. [15, 16] have initiated a comprehensive search for patterns that match the LSM anomaly (or the anomalies of UV systems more generally) with the anomalies of emergent QSL phases, regardless of whether these phases are gapped or gapless. Anomaly matching provides a valuable framework for understanding how crystalline symmetry acts on low-energy degrees of freedom in the infrared limit, ultimately allowing us to systematically classify all possible realizations of these QSL phases on a given lattice.

Focusing on three dimensions, one of the most extensively studied examples is the QSL on the pyrochlore lattice [65]. This lattice, as illustrated in Fig. 2, is proposed to host a QSL phase since the very birth of the concept [66]. A significant theoretical breakthrough occurred in 2004, when Ref. [67] established a mapping from the pyrochlore XXZ model with local Ising anisotropy to a U(1) QSL phase—commonly referred to as the "quantum spin ice" or "Coulomb phase" [68]—described by an emergent Maxwell's theory of electrodynamics. Since then, the properties of pyrochlore QSLs have been the focus of extensive study [69–72], with numerous experiments reporting liquid-like behaviors in rare-earth pyrochlore materials [73–77].

Now we move to the field-theoretic aspects of $(3+1)$-dimensional U(1) Maxwell theory that describes the quantum spin ice phase. The theory has been studied in great detail in this context [78–82]. Here we provide the essential background for later consideration. Generally, a $(3 + 1)$-dimensional U(1) gauge theory is described by the following Lagrangian at low energies:

$$\mathcal{L} = -\frac{1}{4g_0^2} f^{\mu\nu} f_{\mu\nu} + \frac{\theta}{32\pi^2} \epsilon_{\mu\nu\lambda\rho} f^{\mu\nu} f^{\lambda\rho}. \tag{78}$$

Here, $g_0$ is the gauge coupling strength and $\theta$ is the axion angle, which is $2\pi$ or $4\pi$-periodic depending on whether the charges are bosonic or fermionic. At low energies, the theory simply describes propagating photons. Above certain energy threshold, there are fractional excitations carrying electric and magnetic charges, and we denote the excitations carrying one unit of electric (resp. magnetic) charge as $e$ (resp. $m$). For simplicity from here on we restrict to the case where $\theta = 0$.

UV symmetries, including lattice symmetries and internal symmetries, can act on $e$ and $m$ in a nontrivial way. We call it the *symmetry enrichment pattern* of $\mathcal{G}$. The symmetry enrichment pattern can exhibit a rich variety of phenomena [81]:

- Permutation of $e$ and $m$. Specifically, for a unitary symmetry, it either does not change the charges or acts as charge conjugation, i.e., $e \to -e, m \to -m$. For an anti-unitary symmetry, its action on $e$ and $m$ is either $e \to -e, m \to m$ or $e \to e, m \to -m$. In addition, for the reflection $M$ or the inversion $I$, from the Crystalline Equivalence Principle [20] we should treat them as anti-unitary symmetries which also act on $e$ and $m$ as either $e \to -e, m \to m$ or $e \to e, m \to -m$.

  We mention that when both $e$ and $m$ are bosons, the permutation pattern can be highly nontrivial [81]. Yet motivated by [27], we are primarily concerned with the case where $e$ is fermionic while $m$ is bosonic, and thus we do not need to consider them. Note that when both $e$ and $m$ are fermions, the theory has a nontrivial beyond-cohomology gravitational anomaly [78, 83, 84], which is incompatible with the lattice structure we have.

- Symmetry fractionalization of $e$ and $m$. We sometimes also say that $e$ and $m$ carry fractional quantum numbers of the symmetry, which goes beyond a mere sign flip of the charges. There are two pieces of data for symmetry fractionalization: $\omega_e \in H^2_{\rho_e}(\mathcal{G}, \mathrm{U}(1))$ and $\omega_m \in H^2_{\rho_m}(\mathcal{G}, \mathrm{U}(1))$, where the action of $\rho_e$ and $\rho_m$ of $\mathcal{G}$ on $\mathrm{U}(1)$ module is determined by its action on the charges: a group element $g \in \mathcal{G}$ will act on $\mathrm{U}(1)$ module for $e$ by complex conjugation iff it flips the charge of $e$, and similarly for $m$.

Now we connect these field theoretic discussion with parton construction for $\mathrm{U}(1)$ QSLs. In the convention of Ref. [27], a fermionic parton carries both physical spin-1/2 and the charge of the emergent $\mathrm{U}(1)$ gauge field, and we identify it with the electric charge $e$. The magnetic charge $m$ becomes the $\mathrm{U}(1)$ flux as seen by $e$. Nevertheless, it is a nontrivial task to construct the operator corresponding to magnetic charge $m$ [85] within this parton construction.

When $e$ and $m$ are not swapped by the symmetry, one can consider the symmetry fractionalization patterns of individual $e$ (or $m$), and there is a completely general recipe to classify all the symmetry fractionalization patterns for crystalline symmetry, called the PSG classification [86]. Yet it is difficult to obtain the symmetry fractionalization pattern of $e$ and $m$ simultaneously using PSG. Besides, in principle there can be symmetry fractionalization patterns beyond PSG calculation [16]. Later, we will compare our results obtained from LSM anomaly on the 3D pyrochlore lattice with the results obtained from PSG.

With the information above, Ref. [81] proposes that the anomaly for the theory with both $e$ and $m$ bosons is given by

$$\exp\left(-2\pi i\ \omega_e \cup \beta_2(\omega_m)\right). \tag{79}$$

Here $\beta_2$ is the Bockstein homomorphism associated with the short exact sequence $\mathbb{Z} \to \mathbb{R} \to \mathrm{U}(1)$, which maps $\omega_m \in H^2_{\rho_m}(\mathcal{G}, \mathrm{U}(1))$ to $\beta_2(\omega_m) \in H^3_{\rho_m}(\mathcal{G}, \mathbb{Z})$. In the expression (79), $\omega_e \in H^2_{\rho_e}(\mathcal{G}, \mathrm{U}(1))$ takes values in $[0, 1)$. From our requirement of permutation patterns, the total expression is an element in $H^5(\mathcal{G}, \mathrm{U}(1)^{\mathrm{or}})$, as expected.

When $e$ is a fermion and $m$ a boson, according to Ref. [82], an extra $w_2$ term exists in (79), here $w_2$ is the second Stiefel–Whitney class for the tangent bundle of the spacetime manifold $\mathcal{M}_5$. From the Wu formula $Sq^2(\lambda) = (w_2 + w_1^2)\lambda$, $\lambda \in H^3(\mathcal{M}_5, \mathbb{Z}_2)$, we propose that the anomaly for the theory with fermionic $e$ and bosonic $m$ is given by the following expression

$$\exp\left(-2\pi i\ \omega_e \cup \beta_2(\omega_m)\right) \cdot \exp\left(-2\pi i\ \left(Sq^2 + (S\mathcal{Q}^1(1))^2\right)(\beta_2(\omega_m))\right), \tag{80}$$

here the extra $Sq^2$ is the usual Steenrod operation on the group cohomology of $\mathcal{G}$, while $S\mathcal{Q}^1$ is the twisted Steenrod defined in Eq. (38). The total expression is still an element in $H^5(\mathcal{G}, \mathrm{U}(1)^{\mathrm{or}})$. This expression will serve as the IR anomaly $\Omega_{\mathrm{IR}}$ of the system under consideration.

## B. Anomaly matching

In this subsection, we attempt to see what kinds of symmetry fractionalization patterns can indeed match the LSM anomaly with the anomaly Eq. (80) for the emergent $(3+1)$-dimensional $\mathrm{U}(1)$ Maxwell theory (at $\theta = 0$). We will focus on the case where the internal symmetry is $\mathrm{SO}(3)$ and the full symmetry is $\mathcal{G} = G \times \mathrm{SO}(3)$, where $G$ is space group No. 227 ($Fd\bar{3}m$).

As discussed in Sec. II B, the group No. 227 contains four IWPs, $a$, $b$, $c$, and $d$ (see Fig. 2): $c$ and $d$ consist of inversion centers, whereas $a$ and $b$ consist of intersection points of two perpendicular axes. Theses IWPs determine two lattice types: placing physical degrees of freedom on either $c$ or $d$ generates a pyrochlore lattice, and placing them on either $a$ or $b$ generates a diamond lattice.

From the IWP Table of No. 227, we can explicitly write down the LSM anomaly in $H^5(G \times \mathrm{SO}(3), \mathrm{U}(1))$ for putting spin-1/2 degrees of freedom on IWP c (spin-1/2 pyrochlore lattice),

$$\exp\left(i\pi A_i(A_i^2 + A_i A_m + B_{xy+xz+yz})w_2^{\mathrm{SO}(3)}\right). \tag{81}$$

This is the UV anomaly $\Omega_{\mathrm{UV}}$ for our system.

Now we start considering anomaly matching. To be specific, we ask the following question:

*What are all the symmetry fractionalization patterns compatible with anomaly matching?*

As discussed before, we check for a given symmetry action (symmetry fractionalization pattern), whether we can equate the LSM anomaly in Eq. (81) with Eq. (80) or Eq. (79). If the equation cannot hold, then the symmetry fractionalization pattern cannot appear on the given lattice, and the corresponding symmetry-enriched U(1) QSL cannot emerge in this lattice. In contrast, if the equation can hold, we conjecture that the corresponding symmetry-enriched U(1) QSL can emerge in this lattice, according to the hypothesis of emergibility [15, 64].

Because $H^2_{\rho_e}(\mathcal{G}, \mathrm{U}(1)) = H^2_{\rho_e}(G, \mathrm{U}(1)) \oplus H^2(\mathrm{SO}(3), \mathrm{U}(1))$ and similarly for $m$, we can consider $G$ and SO(3) separately. For SO(3), we say that the nontrivial (trivial) element in $H^2(\mathrm{SO}(3), \mathrm{U}(1))$ suggests that $e$ or $m$ carries projective (integer) representation under SO(3). From the parton consideration of Refs. [27, 87], we will assume that $e$ is a fermion and carries projective representation under SO(3). This way the PSG classification in Refs. [27, 87] gives exactly the data about symmetry fractionalization on $e$. In contrast, $m$ is a boson and carries regular representation under integer representation under SO(3).

To consider the lattice symmetry $G$, we need to track how $G$ permutes these excitations. There are four possible actions of the crystalline symmetry on the excitations:

- $I: e \to e, m \to -m, \ M: e \to e, m \to -m$

- $I: e \to e, m \to -m, \ M: e \to -e, m \to m$

- $I: e \to -e, m \to m, \ M: e \to e, m \to -m$

- $I: e \to -e, m \to m, \ M: e \to -e, m \to m$

We will use $(w_I, w_M)$ to label whether the generator $I$ or $M$ gives an extra $-1$ or not when acting on $e$, and the four cases correspond to $(w_I, w_M) = (0,0), (0,1), (1,0), (1,1)$, respectively [17]. We focus on $(w_I, w_M) = (0,0)$, which can match the field theory for quantum spin ice with $\theta = 0$.

Specializing to $(w_I, w_M) = (0,0)$, we can immediately read all the possible symmetry fractionalization classes from Table V. The symmetry fractionalization of the crystalline symmetry $G$ for $e$ is classified by $H^2(G, \mathrm{U}(1)) = H^3(G, \mathbb{Z}) \cong (\mathbb{Z}_2)^3$, where the action of $G$ on U(1) or $\mathbb{Z}$ is trivial. Written in terms of mod-2 cohomology, it is generated by

$$\exp(i\pi B_{xy+xz+yz}), \quad \exp(i\pi B_\alpha), \quad \text{and} \quad \exp(i\pi A_i A_m). \tag{82}$$

In the notation of Ref. [27], the coefficient of the three generators are exactly $\chi_1$, $\chi_{\overline{C}_6 S}$ and $\chi_{ST_1}$ (divided by $\pi$).

The symmetry fractionalization of the crystalline symmetry $G$ for $m$ is classified by $H^2(G, \mathrm{U}(1)^{\mathrm{or}}) = \mathrm{Tor}\left(H^3(G, \mathbb{Z}^{\mathrm{or}})\right) \cong (\mathbb{Z}_2)^4$, where the action of both $M$ and $I$ on U(1) or $\mathbb{Z}$ is nontrivial. Written in terms of mod-2 cohomology, it is generated by

$$\exp(i\pi B_{xy+xz+yz}), \quad \exp(i\pi B_\alpha), \quad \exp(i\pi A_i^2), \quad \text{and} \quad \exp(i\pi A_m^2). \tag{83}$$

This information is not easily obtained from the PSG calculation.

Now we just need to enumerate, for each symmetry fractionalization pattern given by Eqs. (82) and (83), whether the obtained IR anomaly $\Omega_{\mathrm{IR}}$ of the $(3+1)$-dimensional U(1) gauge theory from Eq. (80) is identical to the UV anomaly $\Omega_{\mathrm{UV}}$ from Eq. (81), i.e.,

$$\Omega_{\mathrm{UV}} = \Omega_{\mathrm{IR}}. \tag{84}$$

In the end, we have the following result. First, the symmetry fractionalization of $m$ is fixed to be

$$\exp\left(i\pi(A_i^2 + B_{xy+xz+yz})\right). \tag{85}$$

Physically, this means that, in a U(1) QSL, the magnetic charge $m$ must carry fractional charges of inversion and $T_i T_j T_i^{-1} T_j^{-1} = -1$ when acting on $m$. The symmetry fractionalization of $e$ can be written as

$$\exp\left(i\pi w_2^{\mathrm{SO}(3)} + i\chi_1 B_{xy+xz+yz} + i\chi_{\overline{C}_6 S} B_\alpha\right), \tag{86}$$

---

[17] In the notation of [27], $\overline{C}_6 = C_3 I$, $S = C_2 MI$. Hence, $(w_{\overline{C}_6}, w_S)$ is equal to $(w_I, w_M + w_I \mod 2)$.

where $\chi_1$ and $\chi_{\overline{C}_6 S}$ can take both values $\{0, \pi\}$, and the first term involving $w_2^{\mathrm{SO}(3)}$ signals that $e$ is a spin-1/2 under SO(3).[18]

We conclude that, given the assumption that $e$ is a half-integer-spin fermion that does not change sign under the action of $M$ and $I$, i.e., $(w_I, w_M) = (0, 0)$, and $m$ is an integer-spin boson,

1. The calculation from pure anomaly matching gives exactly the same result as the PSG calculation in Ref. [27].

2. It also gives the symmetry fractionalization of magnetic charges $m$ to be Eq. (85).

It is interesting to compare the PSG calculation for other options of $(w_I, w_M)$ with the calculation of anomaly matching shown above, and obtain further constraints on the emergent IR theory possibly with a nonzero $\theta$-term. We leave it to future work.

## VII.   Summary and discussion

In this paper, we obtained the mod-2 cohomology rings for all 230 3D space groups (except for the high-degree relations of the No. 226 space group) and studied the connection between cohomology elements and real space lattice structures. For a given lattice symmetry $G$, we demonstrated a way to associate each irreducible Wyckoff position–high symmetry points on the lattice roughly speaking–with a unique element in the third group cohomology of $G$, i.e. a 3-cocycle $\lambda \in H^3(G, \mathbb{Z}_2)$, using the notion of topological invariants. We derived this lattice–cohomology (or, more precisely, IWP–3-cocycle) correspondence in the most explicit manner by obtaining the inhomogeneous functions for the 3-cocycles (except for the groups No. 225, 227, and 229).

This lattice–cohomology (or IWP–3-cocycle) correspondence is the essential mathematical description for Lieb–Schultz–Mattis constraints—namely, the lattice constraints that forbid a magnetic crystal consisting of projective degrees of freedom (such as spin-1/2 moments) to have a unique, symmetric, gapped ground state. We assembled the complete set of quantum anomalies that characterize the LSM constraints from the 3-cocycles of the space group cohomology, $\lambda$, and the projective representations of the internal symmetry (assumed to be classified by powers of $\mathbb{Z}_2$). We applied these explicit data of quantum anomaly to the study of $(3+1)$-dimensional U(1) quantum spin liquids on the pyrochlore lattice and compare the results we obtained with projective symmetry group calculations.

We conclude the paper with a few comments and suggestions for future directions.

Obtaining the full structure of group cohomology for 3D space groups is a difficult task. We hope that the techniques and codes that we developed for mod-2 cohomology can be used to calculate other homological algebraic aspects of 3D space groups, including (co)homology and (co)bordism for 3D space groups, and also the equivariant version of them, which are important in understanding the topological phases protected/enriched by these crystalline symmetries. For example, we believe that our result will be important in understanding 3D crystalline SPTs, and could ultimately lead to a comprehensive classification of bosonic and fermionic SPTs protected by 3D space groups, as discussed in Ref. [20] (also see Eq. (95)) and Refs. [23–25].

Another direction is to study the structure of group cohomology for higher dimensional crystallographic groups. Notable dimensions are $k = 4$, where a full classification of crystallographic groups has been achieved, and $k = 6$, which is of special interest to string theory and may inform the study of quasicrystals in 3D (or the fusion of these two subjects [88]). While conceptually this is a straightforward generalization, development on the computational homotopy side awaits.

Turning to the story of LSM constraints, as far as we know, a general proof of the complete LSM constraints in 3D that we discussed in this work is lacking. We would like to see how our results can shed light on a rigorous proof. Even more broadly, we seek to establish a clear connection between general symmetry actions and the elements in group cohomology that classify anomalies. We note that the connection has been rigorously established just recently for the 1D case [56].

Having obtained the cohomology data corresponding to various LSM constraints in 3D, we illustrate the application of our results through a specific example in Sec. VI. We expect that our findings will have significant implications for understanding how different low-energy phases can be realized in specific lattice systems, thereby expanding our knowledge of the complex nature of phase diagrams in real materials with diverse lattice structures. Below, we outline a few candidates.

- We anticipate that our analysis of $(3+1)$-dimensional quantum spin liquids can be generalized to provide a comprehensive understanding of symmetry actions in various U(1) or $\mathbb{Z}_2$ quantum spin liquids across different lattice systems, similar to the $(2+1)$-dimensional story in Ref. [16]. This approach complements the projected symmetry group (PSG) analysis for 3D lattices presented in prior works [27, 87, 89–97]. We believe that employing the

---

[18] When we allow $m$ to carry projective representation under SO(3), there is one more possibility that matches the anomaly of the $(3+1)$-dimensional U(1) gauge theory. Here the symmetry fractionalization of $m$ is fixed to be

$$\exp\left( i\pi(w_2^{\mathrm{SO}(3)} + A_m^2 + B_{xy+xz+yz}) \right),$$

while the symmetry fractionalization of $e$ is also fixed

$$\exp\left( i\pi w_2^{\mathrm{SO}(3)} \right),$$

corresponding to the situation where all three $\chi_1, \chi_{\overline{C}_6 S}, \chi_{ST_1}$ are zero. This possibility seems to be incompatible with the lattice construction in e.g. [67]. And we are curious if it can really realized on the pyrochlore lattice.

method of anomaly matching will yield valuable extra information about how crystalline or UV symmetries act on the emergent degrees of freedom in these systems [98].

- A simple class of phase in these lattice systems is the ferromagnetic phase, or magnetic order in general, which falls under the category of general spontaneous symmetry breaking phase. Even though the ground state of ferromagnets on the lattice can be straightforward, understanding its low energy dynamics, particularly elucidating the detailed matching of the symmetry actions in the lattice and in the continuum can be subtle, as emphasized in [99]. Investigating spontaneous symmetry breaking phases in the lens of anomaly and anomaly matching has proven to provide valuable insights into these systems [100–104], and we expect that applying these methods to crystalline symmetries can yield equally fruitful results.

Finally, it is intriguing to explore how our results about LSM constraints can be generalized to other systems. Notable examples include systems with filling constraints [54, 105–107], systems with magnetic space group symmetries [35, 108–110], systems with long-range interactions [111–113], systems with conserved dipole moments [114, 115], systems with disorders [116], systems with Majorana translations [57–59, 117], open systems [118, 119], and many more [109, 120–125].

## Acknowledgments

We thank Leon Balents, Federico Becca, Andrea Cappelli, Hank Chen, Benoît Douçot, Graham Ellis, Dominic Else, Zheng-Cheng Gu, Zhaoyu Han, Yin-Chen He, Jiankang Huang, Yasir Iqbal, Ethan Lake, Siddhant Mal, Joel Moore, Adam Nahum, Shang-Qiang Ning, Frédéric Piéchon, Andrew C. Potter, Yang Qi, Nathan Seiberg, Ryan Thorngren, Cenke Xu, Xu Yang, Liujun Zou, and many others for helpful discussions. C.L. would like to express deep gratitude to Bill Jacob for the early discussions and guidance that inspired this project. This research was supported in part through computational resources and services provided by Quantum Advanced Research Computing (QuARC) at the Stewart Blusson Quantum Matter Institute. C.L. acknowledges the fellowship support from the Gordon and Betty Moore Foundation through the Emergent Phenomena in Quantum Systems (EPiQS) program.

## Appendix A: Information about point groups and their group cohomology in 3D

In this appendix, we list necessary information about point groups and their group cohomology for 3D space groups. These information is organized in Table I and Table II.

In Table I, we list the basic information about the point groups, including their international (Hermann–Mauguin) notation, Schönflies notation, abstract group structure, order, and generators in "polycyclic order" in permutation notation and notation we use in Appendix F. In the first column we provide the ITC numbering of the space groups associated with each point group, and in the second last column, the numbering of the generators in ITC [32] corresponding to the polycyclic generators.

TABLE I: Information about the structure of the 32 point groups. ∗: the symmetry operation may differ from the ITC operation by a translation. †: the generators (5) and (8) needs to be switched for space groups No. 115–120. ‡: (10) instead of (7) for space groups No. 187–188.

| No. | Point Group Information | | | | | | |
|---|---|---|---|---|---|---|---|
| | Intl | Schönflies | Abstract | Order | Permutation Notation | ITC Numbering* | Polycyclic Gens |
| 1 | 1 | $C_1$ | $\mathbb{Z}_1$ | 1 | Trivial | Trivial | Trivial |
| 2 | $\bar{1}$ | $C_i$ | | | | | $\langle I \rangle$ |
| 3–5 | 2 | $C_2$ | $\mathbb{Z}_2$ | 2 | $\langle (12) \rangle$ | (2) | $\langle C_2 \rangle$ |
| 6–9 | $m$ | $C_s$ | | | | | $\langle M \rangle$ |
| 10–15 | $2/m$ | $C_{2h}$ | | | | | $\langle C_2, I \rangle$ |
| 16–24 | 222 | $D_2$ | $(\mathbb{Z}_2)^2$ | 4 | $\langle (12), (34) \rangle$ | (2),(3) | $\langle C_2, C_2' \rangle$ |
| 25–46 | $mm2$ | $C_{2v}$ | | | | | $\langle C_2, M \rangle$ |
| 47–74 | $mmm$ | $D_{2h}$ | $(\mathbb{Z}_2)^3$ | 8 | $\langle (12), (34), (56) \rangle$ | (2), (3), (5) | $\langle C_2, C_2', I \rangle$ |
| 75–80 | 4 | $C_4$ | $\mathbb{Z}_4$ | 4 | $\langle (1234) \rangle$ | (3) | $\langle C_4 \rangle$ |
| 81–82 | $\bar{4}$ | $S_4$ | | | | | $\langle \overline{C_4} \rangle$ |
| 83–88 | $4/m$ | $C_{4h}$ | $\mathbb{Z}_4 \times \mathbb{Z}_2$ | 8 | $\langle (1234), (56) \rangle$ | (3), (5) | $\langle C_4, I \rangle$ |
| 89–98 | 422 | $D_4$ | | | | | $\langle C_2, C_2', C_2'' \rangle$ |
| 99–110 | $4mm$ | $C_{4v}$ | $Dih_4$ | 8 | $\langle (13)(24), (13)(56), (14)(23)(56) \rangle$ | (2), (5)$^\dagger$, (8)$^\dagger$ | $\langle C_2, M', M \rangle$ |
| 111–122 | $\bar{4}2m$ | $D_{2d}$ | | | | | $\langle C_2, C_2', M \rangle$ |
| 123–142 | $4/mmm$ | $D_{4h}$ | $Dih_4 \times \mathbb{Z}_2$ | 16 | $\langle (13)(24), (13)(56), (14)(23)(56), (78) \rangle$ | (2), (5), (16), (9) | $\langle C_2, C_2', M, I \rangle$ |
| 143–146 | 3 | $C_3$ | $\mathbb{Z}_3$ | 3 | $\langle (123) \rangle$ | (2) | $\langle C_3 \rangle$ |
| 147–148 | $\bar{3}$ | $S_6$ | $\mathbb{Z}_3 \times \mathbb{Z}_2$ | 6 | $\langle (123), (45) \rangle$ | (2), (4) | $\langle C_3, I \rangle$ |
| 149–155 | 32 | $D_3$ | | | | | $\langle C_3, C_2' \rangle$ |
| 156–161 | $3m$ | $C_{3v}$ | $Dih_3$ | 6 | $\langle (123), (23)(45) \rangle$ | (2), (4) | $\langle C_3, M \rangle$ |

| | | | | | | | |
|---|---|---|---|---|---|---|---|
| 162–167 | $\bar{3}m$ | $D_{3d}$ | $Dih_3 \times \mathbb{Z}_2$ | 12 | $\langle(123),(23)(45),(67)\rangle$ | $(2),(4),(7)$ | $\langle C_3, C_2', I\rangle$ |
| 168–173 | 6 | $C_6$ | $\mathbb{Z}_3 \times \mathbb{Z}_2$ | 6 | $\langle(123),(45)\rangle$ | $(2),(4)$ | $\langle C_3, C_2\rangle$ |
| 174 | $\bar{6}$ | $C_{3h}$ | | | | | $\langle C_3, M\rangle$ |
| 175–176 | $6/m$ | $C_{6h}$ | $\mathbb{Z}_3 \times \mathbb{Z}_2^2$ | 12 | $\langle(123),(45),(67)\rangle$ | $(2),(4),(7)$ | $\langle C_3, C_2, I\rangle$ |
| 177–182 | 622 | $D_6$ | $Dih_3 \times \mathbb{Z}_2$ | 12 | $\langle(123),(23)(45),(67)\rangle$ | $(2),(7)[(10)]^{\ddagger},(4)$ | $\langle C_3, C_2', C_2\rangle$ |
| 183–186 | 6mm | $C_{6v}$ | | | | | $\langle C_3, M, C_2\rangle$ |
| 187–190 | $\bar{6}m2$ | $D_{3h}$ | | | | | $\langle C_3, C_2', M\rangle$ |
| 191–194 | 6/mmm | $D_{6h}$ | $Dih_3 \times (\mathbb{Z}_2)^2$ | 24 | $\langle(123),(23)(45),(67),(89)\rangle$ | $(2),(7),(4),(13)$ | $\langle C_3, C_2', C_2, I\rangle$ |
| 195–199 | 23 | $T$ | $A_4$ | 12 | $\langle(12)(34),(13)(24),(321)\rangle$ | $(2),(3),(5)$ | $\langle C_2, C_2', C_3\rangle$ |
| 200–206 | $m\bar{3}$ | $T_h$ | $A_4 \times \mathbb{Z}_2$ | 24 | $\langle(12)(34),(13)(24),(321),(56)\rangle$ | $(2),(3),(5),(13)$ | $\langle C_2, C_2', C_3, I\rangle$ |
| 207–214 | 432 | $O$ | $S_4$ | 24 | $\langle(12)(34),(13)(24),(321),(12)\rangle$ | $(2),(3),(5),(14)$ | $\langle C_2, C_2', C_3, C_2''\rangle$ |
| 215–220 | $\bar{4}3m$ | $T_d$ | | | | $(2),(3),(5),(13)$ | $\langle C_2, C_2', C_3, M\rangle$ |
| 221–230 | $m\bar{3}m$ | $O_h$ | $S_4 \times \mathbb{Z}_2$ | 48 | $\langle(12)(34),(13)(24),(321),(12),(56)\rangle$ | $(2),(3),(5),(38),(25)$ | $\langle C_2, C_2', C_3, M, I\rangle$ |

In Table II, we present information about the group cohomology of these point groups, including their mod-2 cohomology rings, Stiefel–Whitney classes for the natural 3D representation, and topological invariants to detect the associated IWPs in $H^3(G, \mathbb{Z}_2)$.

Notably, point groups with the same abstract group structure—such as $C_i$, $C_2$ and $C_s$, all of which are isomorphic to $\mathbb{Z}_2$—have identical mod-2 cohomology rings. However, they differ by their action on the 3D Euclidean space. The 3D Euclidean space can be viewed as a natural 3D representation of the point group, and we list the first, second, and third Stiefel–Whitney classes for this representation. It is evident that these differing actions are reflected in the distinct Stiefel–Whitney classes associated with the 3D representations.

In the third column about the mod-2 cohomology ring of point groups, the degree-1 generator $A_\bullet$ always denotes (the mod-2 reduction of) the character associated with the group element indicated by the subscript. For example, $A_{c'}$ denotes the character associated with a $C_2'$ rotation.[19]

Finally, in the last column, we list the topological invariants used to identify the corresponding space group cohomology elements in $H^3(G, \mathbb{Z}_2)$, based on the little groups of the IWPs. The definition of $\varphi_1$, $\varphi_2$ and $\varphi_3$ are given in Eqs. (75), (74) and (73), respectively.

If the little group is neither $C_1$ nor $C_s$, the topological invariant associated with an IWP is fully determined by its corresponding little group. For these cases, we list the topological invariant we choose for the given little group. In particular, when $w_3$ is nonzero, the topological invariant just detects $w_3$ in $H^3(P, \mathbb{Z}_2)$, such that the action of $w_3$ on it gives 1. When $w_3$ is zero, the corresponding IWP is a translation unit cell along a $C_2$ rotation axis, and we choose the topological invariant to be $\varphi_2[X, C_2]$, where we use $X$ to label the translation, glide, or screw along the rotation axis.

If the little group is $C_1$ (i.e. trivial), the topological invariant must be that of one of the 10 3D Bieberbach groups, listed in Eq. (61), and we list the topological invariant for each case separately below their IWP tables in Appendix F. When the little group is $C_s$, depending on whether the action of space group in the reflection plane is either isomorphic to $p1$ or $pg$, we choose the topological invariant to be either $\varphi_3[T_1, T_2, M]$ or $\widetilde{\varphi_3}[T_1, G_2, M]$, where

$$\begin{aligned}
\widetilde{\varphi_3}[T_1, G_2, M] =& \lambda(M, T_1 G_2, T_1 G_2^{-1}) + \lambda(M, T_1, G_2) + \lambda(M, T_1, G_2^{-1}) + \lambda(M, G_2, G_2^{-1}) \\
& + \lambda(T_1 G_2, M, T_1 G_2^{-1}) + \lambda(T_1, M, G_2) + \lambda(T_1, M, G_2^{-1}) + \lambda(G_2, M, G_2^{-1}) \\
& + \lambda(T_1 G_2, T_1 G_2^{-1}, M) + \lambda(T_1, G_2, M) + \lambda(T_1, G_2^{-1}, M) + \lambda(G_2, G_2^{-1}, M).
\end{aligned} \tag{87}$$

In the first case, $T_1$ and $T_2$ are two translations along the reflection plane. In the second case, $T_1$, $G_2$ are translation and glide along the reflection plane, respectively, such that $G_2^{-1} T_1 G_2 = T_1^{-1}$.

For little groups that contain a $C_3$ rotation, the topological invariant remains the same as that of the little group after disregarding the $C_3$ rotation. But for the little groups labeled by $(s)$, multiple IWPs may share the same cohomology element, or there may be no corresponding cohomology element at all.

TABLE II: Information about the mod-2 cohomology of the 32 space groups. We defined $\mathcal{R} := B_\alpha^3 + C_{\alpha 1}^2 + C_{\alpha 1} C_{\alpha 2} + C_{\alpha 2}^2$ and $w_3^{47-74} := (A_i + A_c)(A_i + A_{c'})(A_i + A_c + A_{c'})$. In the last column, $X$ represents a translation, glide, or screw along the $C_2$ rotation axis. We use $\star$ to indicate that when the little group for the IWP is $C_1$ or $C_s$, each case must be considered separately. Groups labeled with "(s)" imply that when the little group for the IWP falls into this category, multiple IWPs may share the same cohomology element, or there may be no corresponding cohomology element at all.

| Point Group | | Mod-2 Cohomology Ring Information | | | | Topo. Inv. |
|---|---|---|---|---|---|---|
| Schönflies | Abstract | Expression | $w_1$ | $w_2$ | $w_3$ | for IWP |
| $C_1$ | $\mathbb{Z}_1$ | $\mathbb{Z}_2[]$ | 0 | 0 | 0 | $\star$ |

———

[19] We comment that, for degree-2 and degree-3 generators, although we use the same label (for example, $B_\alpha$) in many point groups, these elements should not be assumed the same. In fact, there is no way to compare them unless there is a subgroup relation. For two point groups $P$ and $P'$ with $P' \subset P$, even if their cohomology contains elements with the same label, we do not require that the generator of the cohomology of $P$ reduces to the one with the same label in the cohomology of $P'$. For example, although $D_{4h} \subset O_h$, and both of their mod-2 cohomology contain the element labeled by $B_\alpha$, $B_\alpha$ of $H^*(O_h, \mathbb{Z}_2)$ actually reduces to $B_\alpha + A_{c'}^2 + A_i A_{c'} + A_i A_m + A_i^2$ (and not $B_\alpha$) of $H^*(D_{4h}, \mathbb{Z}_2)$. Nevertheless, as far as the mod-2 cohomology ring of the *space group* is concerned, we have made sure that, whenever a generator of the mod-2 cohomology of a space group is the pullback of a generator of its associated point group, they are labeled the same.

| | | | | | | |
|---|---|---|---|---|---|---|
| $C_i$ | | $\mathbb{Z}_2[A_i]$ | $A_i$ | $A_i^2$ | $A_i^3$ | $\varphi_1[I]$ |
| $C_2$ | $\mathbb{Z}_2$ | $\mathbb{Z}_2[A_c]$ | 0 | $A_c^2$ | 0 | $\varphi_1[X,C_2]$ |
| $C_s$ | | $\mathbb{Z}_2[A_m]$ | $A_m$ | 0 | 0 | $\star$ |
| $C_{2h}$ | | $\mathbb{Z}_2[A_c,A_i]$ | $A_i$ | $A_i^2+A_c^2$ | $A_i(A_i^2+A_c^2)$ | $\varphi_1[I]$ |
| $D_2$ | $(\mathbb{Z}_2)^2$ | $\mathbb{Z}_2[A_c,A_{c'}]$ | 0 | $A_c^2+A_cA_{c'}+A_{c'}^2$ | $A_cA_{c'}(A_c+A_c')$ | $\varphi_2[C_2,C_2']$ |
| $C_{2v}$ | | $\mathbb{Z}_2[A_c,A_m]$ | $A_m$ | $A_c(A_c+A_m)$ | 0 | $\varphi_1[X,C_2]$ |
| $D_{2h}$ | $(\mathbb{Z}_2)^3$ | $\mathbb{Z}_2[A_c,A_{c'},A_i]$ | $A_i$ | $A_i^2+A_c^2+A_cA_{c'}+A_{c'}^2$ | $w_3^{47-74}$ | $\varphi_2[C_2,C_2']$ |
| $C_4$ | $\mathbb{Z}_4$ | $\mathbb{Z}_2[A_{\mathsf{q}},B_\alpha]/\langle A_{\mathsf{q}}^2\rangle$ | 0 | $B_\alpha$ | 0 | $\varphi_2[X,C_4^2]$ |
| $S_4$ | | $\mathbb{Z}_2[A_{\mathsf{q}},B_\alpha]/\langle A_{\mathsf{q}}^2\rangle$ | $A_{\mathsf{q}}$ | $B_\alpha$ | $A_{\mathsf{q}}B_\alpha$ | $\varphi_2[\overline{C}_4,\overline{C}_4^2]$ |
| $C_{4h}$ | $\mathbb{Z}_4\times\mathbb{Z}_2$ | $\mathbb{Z}_2[A_{\mathsf{q}},A_i,B_\alpha]/\langle A_{\mathsf{q}}^2\rangle$ | $A_i$ | $B_\alpha$ | $A_iB_\alpha$ | $\varphi_1[I]$ |
| $D_4$ | | $\mathbb{Z}_2[A_{c'},A_{c''},B_\alpha]/\langle A_{c'}A_{c''}\rangle$ | 0 | $B_\alpha+A_{c'}^2+A_{c''}^2$ | $(A_{c'}+A_{c''})B_\alpha$ | $\varphi_2[C_2,C_2']$ |
| $C_{4v}$ | $Dih_4$ | $\mathbb{Z}_2[A_{m'},A_m,B_\alpha]/\langle A_{m'}A_m\rangle$ | $A_{m'}+A_m$ | $B_\alpha$ | 0 | $\varphi_2[X,C_2]$ |
| $D_{2d}$ | | $\mathbb{Z}_2[A_{c'},A_m,B_\alpha]/\langle A_{c'}A_m\rangle$ | $A_m$ | $B_\alpha+A_{c'}^2$ | $A_{c'}B_\alpha$ | $\varphi_2[C_2,C_2']$ |
| $D_{4h}$ | $Dih_4\times\mathbb{Z}_2$ | $\mathbb{Z}_2[A_{c'},A_i,A_m,B_\alpha]/\langle A_{c'}A_m\rangle$ | $A_i+A_m$ | $B_\alpha+A_{c'}^2+A_iA_{c'}+A_iA_m$ | $(A_{c'}+A_i)B_\alpha$ | $\varphi_2[C_2,C_2']$ |
| $C_3$ | $\mathbb{Z}_3$ | id. $C_1$ | id. $C_1$ | id. $C_1$ | id. $C_1$ | id. $C_1$ (s) |
| $S_6$ | $\mathbb{Z}_3\times\mathbb{Z}_2$ | id. $C_i$ | id. $C_i$ | id. $C_i$ | id. $C_i$ | id. $C_i$ |
| $D_3$ | $Dih_3$ | id. $C_2$ | id. $C_2$ | id. $C_2$ | id. $C_2$ | id. $C_2$ (s) |
| $C_{3v}$ | | id. $C_s$ | id. $C_s$ | id. $C_s$ | id. $C_s$ | id. $C_s$ (s) |
| $D_{3d}$ | $Dih_3\times\mathbb{Z}_2$ | id. $C_{2h}$ | id. $C_{2h}$ | id. $C_{2h}$ | id. $C_{2h}$ | id. $C_{2h}$ |
| $C_6$ | $\mathbb{Z}_3\times\mathbb{Z}_2$ | id. $C_2$ | id. $C_2$ | id. $C_2$ | id. $C_2$ | id. $C_2$ (s) |
| $C_{3h}$ | | id. $C_s$ | id. $C_s$ | id. $C_s$ | id. $C_s$ | id. $C_s$ (s) |
| $C_{6h}$ | $\mathbb{Z}_3\times\mathbb{Z}_2^2$ | id. $C_{2h}$ | id. $C_{2h}$ | id. $C_{2h}$ | id. $C_{2h}$ | id. $C_{2h}$ |
| $D_6$ | | id. $D_2$ | id. $D_2$ | id. $D_2$ | id. $D_2$ | id. $D_2$ |
| $C_{6v}$ | $Dih_3\times\mathbb{Z}_2$ | id. $C_{2v}$ | id. $C_{2v}$ | id. $C_{2v}$ | id. $C_{2v}$ | id. $C_{2v}$ (s) |
| $D_{3h}$ | | id. $C_{2v}$ | id. $C_{2v}$ | id. $C_{2v}$ | id. $C_{2v}$ | id. $C_{2v}$ (s) |
| $D_{6h}$ | $Dih_3\times(\mathbb{Z}_2)^2$ | id. $D_{2h}$ | id. $D_{2h}$ | id. $D_{2h}$ | id. $D_{2h}$ | id. $D_{2h}$ |
| $T$ | $A_4$ | $\mathbb{Z}_2[B_\alpha,C_{\alpha1},C_{\alpha2}]/\langle\mathcal{R}\rangle$ | 0 | $B_\alpha$ | $C_{\alpha1}+C_{\alpha2}$ | id. $D_2$ |
| $T_h$ | $A_4\times\mathbb{Z}_2$ | $\mathbb{Z}_2[A_i,B_\alpha,C_{\alpha1},C_{\alpha2}]/\langle\mathcal{R}\rangle$ | $A_i$ | $B_\alpha+A_i^2$ | $C_{\alpha1}+C_{\alpha2}+A_iB_\alpha+A_i^3$ | id. $D_{2h}$ |
| $O$ | $S_4$ | $\mathbb{Z}_2[A_{c''},B_\alpha,C_\alpha]/\langle A_{c''}C_\alpha\rangle$ | 0 | $B_\alpha+A_{c''}^2$ | $C_\alpha+A_{c''}B_\alpha$ | id. $D_4$ |
| $T_d$ | | $\mathbb{Z}_2[A_m,B_\alpha,C_\alpha]/\langle A_mC_\alpha\rangle$ | $A_m$ | $B_\alpha$ | $C_\alpha$ | id. $D_{2d}$ |
| $O_h$ | $S_4\times\mathbb{Z}_2$ | $\mathbb{Z}_2[A_i,A_m,B_\alpha,C_\alpha]/\langle A_mC_\alpha\rangle$ | $A_i+A_m$ | $B_\alpha+A_i^2$ | $C_\alpha+A_iB_\alpha+A_i^2(A_i+A_m)$ | id. $D_{4h}$ |

## Appendix B: Information about space groups as group extensions

In this appendix, we summarize how to view the space groups $G$ as its point group $P$ extended by the translation group $T\cong\mathbb{Z}^3$ (see Eq. (1)). These properties are listed in Table III.

As introduced in the Sec. II A, given a point group $P$, the conjugation action $\rho$ of $P$ on the translations $T$ is different depending on different Bravais lattice types. The group extension is characterized by the conjugation action $\rho$ of the point group $P$ on translations $T$ (Eq. (2)) and, for a given $\rho$, the second cohomology group $H^2_\rho(P,T)$. For the 32 point groups, there are altogether 14 Bravais lattice types. The distinct actions $\rho$ across all the point groups give rise to 73 *arithmetic crystal classes*.

The 14 Bravais lattice types are $aP,mP,mS,oP,oS,oI,oF,tP,tI,hR,hP,cP,cI,cF$. The first letter is dependent on the point group, and the second letter labels different basis for the translation generators. We choose the generators of translation symmetry according to the following convention:

- $P$ (Primitive) lattices: the three translation generators are

$$T_1: (x,y,z)\to(x+1,y,z), \tag{88a}$$
$$T_2: (x,y,z)\to(x,y+1,z), \tag{88b}$$
$$T_3: (x,y,z)\to(x,y,z+1). \tag{88c}$$

- $S$ (Base-centered) lattices: it has three subtypes $A,B,C$. For $C$ lattices, the three translation generators are

$$T_1: (x,y,z)\to(x+1/2,y+1/2,z), \tag{89a}$$
$$T_2: (x,y,z)\to(x-1/2,y+1/2,z), \tag{89b}$$
$$T_3: (x,y,z)\to(x,y,z+1). \tag{89c}$$

Similarly, for $A$ lattices, the three translation generators are

$$T_1 \colon (x, y, z) \to (x + 1, y, z), \tag{90a}$$
$$T_2 \colon (x, y, z) \to (x, y + 1/2, z + 1/2), \tag{90b}$$
$$T_3 \colon (x, y, z) \to (x, y - 1/2, z + 1/2). \tag{90c}$$

- $I$ (Body-centered) lattices: the three translation generators are

$$T_1 \colon (x, y, z) \to (x - 1/2, y + 1/2, z + 1/2), \tag{91a}$$
$$T_2 \colon (x, y, z) \to (x + 1/2, y - 1/2, z + 1/2), \tag{91b}$$
$$T_3 \colon (x, y, z) \to (x + 1/2, y + 1/2, z - 1/2). \tag{91c}$$

- $F$ (Face-centered) lattices: the three translation generators are

$$T_1 \colon (x, y, z) \to (x, y + 1/2, z + 1/2), \tag{92a}$$
$$T_2 \colon (x, y, z) \to (x + 1/2, y, z + 1/2), \tag{92b}$$
$$T_3 \colon (x, y, z) \to (x + 1/2, y + 1/2, z). \tag{92c}$$

- $R$ (Rhombohedral) lattices: the three translation generators are

$$T_1 \colon (x, y, z) \to (x + 1, y, z), \tag{93a}$$
$$T_2 \colon (x, y, z) \to (x, y + 1, z), \tag{93b}$$
$$T_3 \colon (x, y, z) \to (x + 2/3, y + 1/3, z + 1/3). \tag{93c}$$

The space groups belonging to each arithmetic crystal class are given in the fourth column of Table III [20]. In the fifth column we give the second cohomology group $H^2_\rho(P, T)$, whose elements determine the space groups in the column next to it on the left. In fact, there is the so-called *main theorem of mathematical crystallography*.

**Theorem 19.** *(Main theorem of mathematical crystallography [126, 127]) For k-dimensional crystallography groups with arithmetic crystal class determined by point group $P$ and an integral representation $\rho$ on $T \cong \mathbb{Z}^k$, consider the action of $GL(k, \mathbb{Z})$ on $H^2_\rho(P, T)$, such that given a function $\omega$ representing $[\omega] \in H^2_\rho(P, T)$, we have*

$$(g.\omega)(p_1, p_2) = g.(\omega(g^{-1} p_1 g, g^{-1} p_2 g)), \quad g \in GL(k), \ p_{1,2} \in P. \tag{94}$$

*There exists a one-to-one correspondence between space groups in the arithmetic class $(P, \rho)$ and the orbits of the action in $H^2_\rho(P, T)$.*

We tabulate the results for the 3D space groups in Table III as we have not found them elsewhere. We note that the results for the 2D wallpaper groups are nicely summarized by Morandi [128].

Furthermore, in the last three columns of Table III we give the $\mathbb{Z}_2$ rank of cohomology $H^p(P, H^q(T, \mathbb{Z}_2))$ associated with the arithmetic crystal classes. They appear in the $(p, q)$ entry of the $E_2$ page of the standard LHS spectral sequence (11).

TABLE III: Information on the group extension of the point group $P$ by translations $T = \mathbb{Z}^3$. $\rho$ denotes the action of the point group $P$ on the translation $T$, and $\rho^*$ denotes its induced action of $P$ on the cohomology $H^{1,2}(\mathbb{Z}^3, \mathbb{Z}_2)$. †: this lattice type corresponds to two arithmetic crystal classes. $a_n = (n - 1) - 2\lfloor (n - 1)/3 \rfloor$, $s_n = \lfloor 2n/3 \rfloor + 1$, where the floor function $\lfloor n \rfloor$ gives the integer equal to or less than $n$.

| Point Group | | Lattice | Arithmetic Crystal Class $\rho$ | | Cohomology Dimension in $\mathbb{Z}_2$ | | |
|---|---|---|---|---|---|---|---|
| Abstract | Schönflies | Type | Space Group No. | $H^2_\rho(P, T)$ | $H^n(P, \mathbb{Z}_2)$ | $H^n_{\rho^*}(P, H^1(T, \mathbb{Z}_2))$ | $H^n_{\rho^*}(P, H^2(T, \mathbb{Z}_2))$ |
| Trivial | $C_1$ | $P$ | 1 | Trivial | $\delta_{n,0}$ | $3\delta_{n,0}$ | $3\delta_{n,0}$ |
| | $C_i$ | $P$ | 2 | Trivial | | 3 | 3 |
| $\mathbb{Z}_2$ | $C_2$ | $P$ | 3–4 | $\mathbb{Z}_2$ | 1 | 3 | 3 |
| | | $C$ | 5 | Trivial | | $\delta_{n,0} + 1$ | $\delta_{n=0} + 1$ |
| | $C_s$ | $P$ | 6–7 | $(\mathbb{Z}_2)^2$ | | 3 | 3 |
| | | $C$ | 8–9 | $\mathbb{Z}_2$ | | $\delta_{n,0} + 1$ | $\delta_{n=0} + 1$ |

---

[20] Note that the first group in each row of the column "Arithmetic Crystal Class $\rho$" is not necessarily the symmorphic (i.e. split) one.

| | | | | | | | |
|---|---|---|---|---|---|---|---|
| $\mathbb{Z}_2^2$ | $C_{2h}$ | $P$ | 10–11,13–14 | $(\mathbb{Z}_2)^3$ | $n+1$ | $3(1+n)$ | $3(1+n)$ |
| | | $C$ | 12,15 | $\mathbb{Z}_2$ | | $n+2$ | $n+2$ |
| | $D_2$ | $P$ | 16–19 | $(\mathbb{Z}_2)^3$ | | $3(n+1)$ | $3(n+1)$ |
| | | $C$ | 20–21 | $\mathbb{Z}_2$ | | $n+2$ | $n+2$ |
| | | $F$ | 22 | Trivial | | $n+2$ | $n+\delta_{n,0}$ |
| | | $I$ | 23–24 | $\mathbb{Z}_2$ | | $n+\delta_{n,0}$ | $n+2$ |
| | $C_{2v}$ | $P$ | 25–34 | $(\mathbb{Z}_2)^4$ | | $3(n+1)$ | $3(n+1)$ |
| | | $C$ | 35–37 | $(\mathbb{Z}_2)^2$ | | $n+2$ | $n+2$ |
| | | $A$ | 38–41 | $(\mathbb{Z}_2)^2$ | | $n+2$ | $n+2$ |
| | | $F$ | 42–43 | $\mathbb{Z}_2$ | | $n+2$ | $n+\delta_{n=0}$ |
| | | $I$ | 44–46 | $(\mathbb{Z}_2)^2$ | | $n+\delta_{n,0}$ | $n+2$ |
| $\mathbb{Z}_2^3$ | $D_{2h}$ | $P$ | 47–62 | $(\mathbb{Z}_2)^6$ | $\frac{1}{2}(n+1)(n+2)$ | $\frac{3}{2}(n+1)(n+2)$ | $\frac{3}{2}(n+1)(n+2)$ |
| | | $C$ | 63–68 | $(\mathbb{Z}_2)^3$ | | $\frac{1}{2}(n+1)(n+4)$ | $\frac{1}{2}(n+1)(n+4)$ |
| | | $F$ | 69–70 | $\mathbb{Z}_2$ | | $\frac{1}{2}(n+1)(n+4)$ | $\frac{1}{2}n(n+1)+1$ |
| | | $I$ | 71–74 | $(\mathbb{Z}_2)^3$ | | $\frac{1}{2}n(n+1)+1$ | $\frac{1}{2}(n+1)(n+4)$ |
| $\mathbb{Z}_4$ | $C_4$ | $P$ | 75–78 | $\mathbb{Z}_4$ | $1$ | $2$ | $2$ |
| | | $I$ | 79–80 | $\mathbb{Z}_2$ | | $1$ | $1$ |
| | $S_4$ | $P$ | 81 | Trivial | | $2$ | $2$ |
| | | $I$ | 82 | | | $1$ | $1$ |
| $\mathbb{Z}_4 \times \mathbb{Z}_2$ | $C_{4h}$ | $P$ | 83–86 | $(\mathbb{Z}_2)^2$ | $n+1$ | $2(n+1)$ | $2(n+1)$ |
| | | $I$ | 87–88 | $\mathbb{Z}_2$ | | $n+1$ | $n+1$ |
| $Dih_4$ | $D_4$ | $P$ | 89–96 | $\mathbb{Z}_2 \oplus \mathbb{Z}_4$ | | $2(n+1)$ | $2(n+1)$ |
| | | $I$ | 97–98 | $\mathbb{Z}_2$ | | $n+1$ | $n+1$ |
| | $C_{4v}$ | $P$ | 99–106 | $(\mathbb{Z}_2)^3$ | | $2(n+1)$ | $2(n+1)$ |
| | | $I$ | 107–110 | $(\mathbb{Z}_2)^2$ | | $n+1$ | $n+1$ |
| | $D_{2d}$ | $P^\dagger$ | 111–114 115–118 | $(\mathbb{Z}_2)^2$ | | $2(n+1)$ | $2(n+1)$ |
| | | $I^\dagger$ | 119–120 121–122 | $\mathbb{Z}_2$ | | $n+1$ | $n+1$ |
| $Dih_4 \times \mathbb{Z}_2$ | $D_{4h}$ | $P$ | 123–138 | $(\mathbb{Z}_2)^4$ | $\frac{1}{2}(n+1)(n+2)$ | $(n+1)(n+2)$ | $(n+1)(n+2)$ |
| | | $I$ | 139–142 | $(\mathbb{Z}_2)^2$ | | $\frac{1}{2}(n+1)(n+2)$ | $\frac{1}{2}(n+1)(n+2)$ |
| $\mathbb{Z}_3$ | $C_3$ | $P$ | 143–145 | $\mathbb{Z}_3$ | $\delta_{n,0}$ | $\delta_{n,0}$ | $\delta_{n,0}$ |
| | | $R$ | 146 | Trivial | | | |
| $\mathbb{Z}_3 \times \mathbb{Z}_2$ | $S_6$ | $P$ | 147 | Trivial | $1$ | $1$ | $1$ |
| | | $R$ | 148 | Trivial | | | |
| | $C_6$ | $P$ | 168–173 | $\mathbb{Z}_3 \oplus \mathbb{Z}_2$ | | | |
| | $C_{3h}$ | $P$ | 174 | Trivial | | | |
| $\mathbb{Z}_3 \times \mathbb{Z}_2^2$ | $C_{6h}$ | $P$ | 175–176 | $\mathbb{Z}_2$ | $n+1$ | $n+1$ | $n+1$ |
| $Dih_3$ | $D_3$ | $P^\dagger$ | 149,151,153 150,152,154 | $\mathbb{Z}_3$ | $1$ | $1$ | $1$ |
| | | $R$ | 155 | Trivial | | | |
| | $C_{3v}$ | $P^\dagger$ | 156,158 157,159 | $\mathbb{Z}_2$ | | | |
| | | $R$ | 160,161 | $\mathbb{Z}_2$ | | | |
| $Dih_3 \times \mathbb{Z}_2$ | $D_{3d}$ | $P^\dagger$ | 162–163 164–165 | $\mathbb{Z}_2$ | $n+1$ | $n+1$ | $n+1$ |
| | | $R$ | 166–167 | $\mathbb{Z}_2$ | | | |
| | $D_6$ | $P$ | 177–182 | $\mathbb{Z}_3 \oplus \mathbb{Z}_2$ | | | |
| | $C_{6v}$ | $P$ | 183–186 | $(\mathbb{Z}_2)^2$ | | | |
| | $D_{3h}$ | $P^\dagger$ | 187–188 189–190 | $\mathbb{Z}_2$ | | | |
| $Dih_3 \times \mathbb{Z}_2^2$ | $D_{6h}$ | $P$ | 191–194 | $(\mathbb{Z}_2)^2$ | $\frac{1}{2}(n+1)(n+2)$ | $\frac{1}{2}(n+1)(n+2)$ | $\frac{1}{2}(n+1)(n+2)$ |
| $A_4$ | $T$ | $P$ | 195,198 | $\mathbb{Z}_2$ | $a_n$ | $n+1$ | $n+1$ |
| | | $F$ | 196 | Trivial | | $a_{n+1}$ | $a_{n-1}+\delta_{n,0}$ |
| | | $I$ | 197,199 | $\mathbb{Z}_2$ | | $a_{n-1}+\delta_{n,0}$ | $a_{n+1}$ |
| $A_4 \times \mathbb{Z}_2$ | $T_h$ | $P$ | 200–201,205 | $(\mathbb{Z}_2)^2$ | $\sum_{j=0}^n a_j$ | $\frac{1}{2}(n+1)(n+2)$ | $\frac{1}{2}(n+1)(n+2)$ |
| | | $F$ | 202–203 | $\mathbb{Z}_2$ | | $\sum_{j=0}^n a_{j+1}$ | $1+\sum_{j=0}^n a_{j-1}$ |
| | | $I$ | 204,206 | $\mathbb{Z}_2$ | | $1+\sum_{j=0}^n a_{j-1}$ | $\sum_{j=0}^n a_{j+1}$ |
| $S_4$ | $O$ | $P$ | 207–208,212–213 | $\mathbb{Z}_4$ | $s_n$ | $n+1$ | $n+1$ |
| | | $F$ | 209–210 | $\mathbb{Z}_2$ | | $s_{n-1}+1$ | $s_n+1$ |
| | | $I$ | 211,214 | $\mathbb{Z}_2$ | | $s_n+1$ | $s_{n-1}+1$ |
| | $T_d$ | $P$ | 215,218 | $\mathbb{Z}_2$ | | $n+1$ | $n+1$ |
| | | $F$ | 216,219 | $\mathbb{Z}_2$ | | $s_{n-1}+1$ | $s_n+1$ |
| | | $I$ | 217,220 | $\mathbb{Z}_2$ | | $s_n+1$ | $s_{n-1}+1$ |
| $S_4 \times \mathbb{Z}_2$ | $O_h$ | $P$ | 221–224 | $(\mathbb{Z}_2)^2$ | $\sum_{j=0}^n s_j$ | $\frac{1}{2}(n+1)(n+2)$ | $\frac{1}{2}(n+1)(n+2)$ |

| | | $F$ | 225–228 | $(\mathbb{Z}_2)^2$ | | $1+n+\sum_{j=0}^n s_{j-1}$ | $1+n+\sum_{j=0}^n s_j$ |
| | | $I$ | 229–230 | $\mathbb{Z}_2$ | | $1+n+\sum_{j=0}^n s_j$ | $1+n+\sum_{j=0}^n s_{j-1}$ |

### Appendix C: Information about the cohomology of space groups

In Table IV, we list the basic information about the mod-2 cohomology of all 230 space groups. In Table V, we list the integral cohomology of all 230 space groups.

Specifically, for the mod-2 cohomology, the $\mathbb{Z}_2$ rank in each degree can be summarized by a *Poincaré series*. By Taylor expanding the series around $x = 0$, the $\mathbb{Z}_2$ rank for each degree $n$ can be extracted from the coefficient in front of $x^n$. In the fourth column we list the Poincaré series for each space group.

From this we can immediately assess how fast the $\mathbb{Z}_2$ rank grows with respect to $n$: write the Poincaré series as $P(x)/Q(x)$ where $P(x)$ and $Q(x)$ are integer polynomials of $x$, the $\mathbb{Z}_2$ rank grows as fast as $n^{d-1}$ with respect to $n$ where $d$ is the degree of the integer polynomial $Q(x)$. This $d$ is exactly the *Krull dimensin* of the graded ring $H^*(G, \mathbb{Z}_2)$ [38]. Specially, when $Q(x) = 1$ and the Poincaré series itself is an integer polynomial, the $\mathbb{Z}_2$ rank will be zero for sufficiently large $n$, and this happens for Bieberbach groups. When $\deg Q(x) = 1$, the $\mathbb{Z}_2$ rank will be a constant for sufficiently large $n$. We discuss the two phenomena in Sec. IV B.

In the fifth column, we explicitly list the dimension in the first six degrees, and use a superscript to label the number of the mod-2 cohomology ring generators at this degree (if nonzero).

Moreover, in the third column, we list three features of the group: a "⋊" means the group is symmorphic (i.e. the group splits as point group extended by translation), a "●" means that the standard LHS spectral sequence (11) of this group collapses at the $E_2$ page, and a "♭" means the group is a Bieberbach group.

In Table V, we analyze both trivial and orientation-reversing action of space groups $G$ on $\mathbb{Z}$. In the orientation-reversing action, reflection $M$ and inversion $I$ acts on $\mathbb{Z}$ by multiplication of $-1$, and we use a superscript $^{\mathrm{or}}$ to indicate this nontrivial action. The fifth twisted cohomology, $H^5(G, \mathbb{Z}^{\mathrm{or}})$ listed in the $n = 5$ column of orientation-reversing case agrees with the so-called classification of in-cohomology bosonic SPT protected by crystalline symmetries given in Ref. [20] for all 227 groups that they obtained, following the fact that $H^5(G, \mathbb{Z}^{\mathrm{or}}) \cong H^4(G, \mathrm{U}(1)^{\mathrm{or}})$. Notably, the results for the rest three groups—No. 227, 228, and 230—are missing in Ref. [20], where we give their classification results to be

$$H^5(\text{No. }227, \mathbb{Z}^{\mathrm{or}}) = \mathbb{Z}_2^7, \quad H^5(\text{No. }228, \mathbb{Z}^{\mathrm{or}}) = \mathbb{Z}_2^3, \quad \text{and } H^5(\text{No. }230, \mathbb{Z}^{\mathrm{or}}) = \mathbb{Z}_2^4. \tag{95}$$

TABLE IV: Additional information about space groups and their mod-2 cohomology. The "Feat." column lists the features of the group: "⋊", "●", and "♭" mean the group "is symmorphic", "has a standard LHS spectral sequence that collapses at the $E_2$ page", and "is a Bieberbach group", respectively. $\mathscr{P}(x)$: generating function of the Poincaré series for the $\mathbb{Z}_2$ rank of the mod-2 cohomology $H^n(G, \mathbb{Z}_2)$. The ranks for the first degrees ($n = 1$–6) are listed in the last column, with the number of the mod-2 cohomology ring generators at this degree (if there are any) written as a superscript.

| No. | Name | Feat. | $\mathscr{P}(x)$ | Mod-2 Coh. Dim.$^{\#\text{Gens}}$ | No. | Name | Feat. | $\mathscr{P}(x)$ | Mod-2 Coh. Dim.$^{\#\text{Gens}}$ |
|---|---|---|---|---|---|---|---|---|---|
| 1 | $P1$ | ⋊ ● ♭ | $(x+1)^3$ | $3^3, 3, 1, 0, 0, 0, ...$ | 116 | $P\bar{4}c2$ | | $\frac{-x^3+x^2+x+1}{(x-1)^2}$ | $3^3, 6^3, 8, 10, 12, 14, ...$ |
| 2 | $P\bar{1}$ | ⋊ ● | $-\frac{(x+1)^3}{x-1}$ | $4^4, 7, 8, 8, 8, 8, ...$ | 117 | $P\bar{4}b2$ | | $\frac{x+1}{(x-1)^2}$ | $3^3, 5^2, 7^1, 9, 11, 13, ...$ |
| 3 | $P2$ | ⋊ ● | $-\frac{(x+1)^3}{x-1}$ | $4^4, 7, 8, 8, 8, 8, ...$ | 118 | $P\bar{4}n2$ | | $\frac{x+1}{(x-1)^2}$ | $3^3, 5^1, 7^1, 9, 11, 13, ...$ |
| 4 | $P2_1$ | ♭ | $(x+1)^3$ | $3^3, 3, 1, 0, 0, 0, ...$ | 119 | $I\bar{4}m2$ | ⋊ ● | $\frac{x^3+x^2+x+1}{(x-1)^2}$ | $3^3, 6^3, 10^2, 14, 18, 22, ...$ |
| 5 | $C2$ | ⋊ ● | $-\frac{(x+1)^2}{x-1}$ | $3^3, 4^1, 4, 4, 4, 4, ...$ | 120 | $I\bar{4}c2$ | | $\frac{x^4-x^3+x+1}{(x-1)^2}$ | $3^3, 5^2, 6, 8^2, 10, 12, ...$ |
| 6 | $Pm$ | ⋊ ● | $-\frac{(x+1)^3}{x-1}$ | $4^4, 7, 8, 8, 8, 8, ...$ | 121 | $I\bar{4}2m$ | ⋊ | $\frac{x^2+x+1}{(x-1)^2}$ | $3^3, 6^3, 9^1, 12, 15, 18, ...$ |
| 7 | $Pc$ | ♭ | $(x+1)^3$ | $3^3, 3, 1, 0, 0, 0, ...$ | 122 | $I\bar{4}2d$ | | $\frac{x^3+x^2+x+1}{1-x}$ | $2^2, 3^2, 4^1, 4, 4, 4, ...$ |
| 8 | $Cm$ | ⋊ ● | $-\frac{(x+1)^2}{x-1}$ | $3^3, 4^1, 4, 4, 4, 4, ...$ | 123 | $P4/mmm$ | ⋊ ● | $-\frac{x^3+2x^2+2x+1}{(x-1)^3}$ | $5^5, 14^2, 29, 50, 77, 110, ...$ |
| 9 | $Cc$ | ♭ | $x^3+2x^2+2x+1$ | $2^2, 2^2, 1, 0, 0, 0, ...$ | 124 | $P4/mcc$ | | $\frac{x^3+2x^2+2x+1}{(x-1)^2}$ | $4^4, 9^2, 15, 21, 27, 33, ...$ |
| 10 | $P2/m$ | ⋊ ● | $\frac{(x+1)^3}{(x-1)^2}$ | $5^5, 12, 20, 28, 36, 44, ...$ | 125 | $P4/nbm$ | | $\frac{x^3+2x^2+2x+1}{(x-1)^2}$ | $4^4, 9^2, 15, 21, 27, 33, ...$ |
| 11 | $P2_1/m$ | | $-\frac{(x+1)^3}{x-1}$ | $4^4, 7, 8, 8, 8, 8, ...$ | 126 | $P4/nnc$ | | $\frac{x^2+x+1}{(x-1)^2}$ | $3^3, 6^3, 9^1, 12, 15, 18, ...$ |
| 12 | $C2/m$ | ⋊ ● | $\frac{(x+1)^2}{(x-1)^2}$ | $4^4, 8^1, 12, 16, 20, 24, ...$ | 127 | $P4/mbm$ | | $-\frac{x+1}{(x-1)^3}$ | $4^4, 9^2, 16^1, 25, 36, 49, ...$ |
| 13 | $P2/c$ | | $-\frac{(x+1)^3}{x-1}$ | $4^4, 7, 8, 8, 8, 8, ...$ | 128 | $P4/mnc$ | | $\frac{x^3+x^2+x+1}{(x-1)^2}$ | $3^3, 6^3, 10^2, 14, 18, 22, ...$ |
| 14 | $P2_1/c$ | | $-\frac{(x+1)^2}{x-1}$ | $3^3, 4^1, 4, 4, 4, 4, ...$ | 129 | $P4/nmm$ | | $\frac{x^3+2x^2+2x+1}{(x-1)^2}$ | $4^4, 9^2, 15, 21, 27, 33, ...$ |
| 15 | $C2/c$ | | $\frac{x^3+2x^2+2x+1}{1-x}$ | $3^3, 5^2, 6, 6, 6, 6, ...$ | 130 | $P4/ncc$ | | $\frac{-x^3+x+1}{(x-1)^2}$ | $3^3, 5^2, 6, 7^1, 8, 9, ...$ |
| 16 | $P222$ | ⋊ ● | $\frac{(x+1)^3}{(x-1)^2}$ | $5^5, 12, 20, 28, 36, 44, ...$ | 131 | $P4_2/mmc$ | | $-\frac{2x^2+x+1}{(x-1)^3}$ | $4^4, 11^4, 22, 37, 56, 79, ...$ |
| 17 | $P222_1$ | | $-\frac{(x+1)^3}{x-1}$ | $4^4, 7, 8, 8, 8, 8, ...$ | 132 | $P4_2/mcm$ | | $-\frac{-x^3+x^2+x+1}{(x-1)^3}$ | $4^4, 10^3, 18, 28, 40, 54, ...$ |
| 18 | $P2_12_12$ | | $-\frac{(x+1)^2}{x-1}$ | $3^3, 4^1, 4, 4, 4, 4, ...$ | 133 | $P4_2/nbc$ | | $\frac{x^2+x+1}{(x-1)^2}$ | $3^3, 6^3, 9^1, 12, 15, 18, ...$ |

| No. | Symbol | Marks | Polynomial | Sequence |
|---|---|---|---|---|
| 19 | $P2_12_12_1$ | ♭ | $x^3+2x^2+2x+1$ | $2^2, 2^2, 1, 0, 0, 0, ...$ |
| 20 | $C222_1$ | | $-\frac{(x+1)^2}{x-1}$ | $3^3, 4^1, 4, 4, 4, 4, ...$ |
| 21 | $C222$ | ⋊ ◐ | $\frac{(x+1)^2}{(x-1)^2}$ | $4^4, 8^1, 12, 16, 20, 24, ...$ |
| 22 | $F222$ | ⋊ ◐ | $\frac{x^4+2x+1}{(x-1)^2}$ | $4^4, 7, 10^2, 14, 18, 22, ...$ |
| 23 | $I222$ | ⋊ ◐ | $\frac{x^3+x^2+x+1}{(x-1)^2}$ | $3^3, 6^3, 10^1, 14, 18, 22, ...$ |
| 24 | $I2_12_12_1$ | | $\frac{x^3+2x^2+2x+1}{1-x}$ | $3^3, 5^2, 6, 6, 6, 6, ...$ |
| 25 | $Pmm2$ | ⋊ ◐ | $\frac{(x+1)^3}{(x-1)^2}$ | $5^5, 12, 20, 28, 36, 44, ...$ |
| 26 | $Pmc2_1$ | | $-\frac{(x+1)^3}{x-1}$ | $4^4, 7, 8, 8, 8, 8, ...$ |
| 27 | $Pcc2$ | | $-\frac{(x+1)^3}{x-1}$ | $4^4, 7, 8, 8, 8, 8, ...$ |
| 28 | $Pma2$ | | $-\frac{(x+1)^3}{x-1}$ | $4^4, 7, 8, 8, 8, 8, ...$ |
| 29 | $Pca2_1$ | ♭ | $(x+1)^3$ | $3^3, 3, 1, 0, 0, 0, ...$ |
| 30 | $Pnc2$ | | $-\frac{(x+1)^2}{x-1}$ | $3^3, 4^1, 4, 4, 4, 4, ...$ |
| 31 | $Pmn2_1$ | | $-\frac{(x+1)^2}{x-1}$ | $3^3, 4^1, 4, 4, 4, 4, ...$ |
| 32 | $Pba2$ | | $-\frac{(x+1)^2}{x-1}$ | $3^3, 4^1, 4, 4, 4, 4, ...$ |
| 33 | $Pna2_1$ | ♭ | $x^3+2x^2+2x+1$ | $2^2, 2^2, 1, 0, 0, 0, ...$ |
| 34 | $Pnn2$ | | $-\frac{(x+1)^2}{x-1}$ | $3^3, 4, 4, 4, 4, 4, ...$ |
| 35 | $Cmm2$ | ⋊ ◐ | $\frac{(x+1)^2}{(x-1)^2}$ | $4^4, 8^1, 12, 16, 20, 24, ...$ |
| 36 | $Cmc2_1$ | | $-\frac{(x+1)^2}{x-1}$ | $3^3, 4^1, 4, 4, 4, 4, ...$ |
| 37 | $Ccc2$ | | $\frac{x^3+2x^2+2x+1}{1-x}$ | $3^3, 5^2, 6, 6, 6, 6, ...$ |
| 38 | $Amm2$ | ⋊ ◐ | $\frac{(x+1)^2}{(x-1)^2}$ | $4^4, 8^1, 12, 16, 20, 24, ...$ |
| 39 | $Aem2$ | | $-\frac{(x+1)^3}{x-1}$ | $4^4, 7, 8, 8, 8, 8, ...$ |
| 40 | $Ama2$ | | $\frac{x^3+2x^2+2x+1}{1-x}$ | $3^3, 5^2, 6, 6, 6, 6, ...$ |
| 41 | $Aea2$ | | $\frac{x^3-2x-1}{x-1}$ | $3^3, 3, 2^1, 2, 2, 2, ...$ |
| 42 | $Fmm2$ | ⋊ | $\frac{-x^3+2x+1}{(x-1)^2}$ | $4^4, 7, 9^1, 11, 13, 15, ...$ |
| 43 | $Fdd2$ | | $\frac{x+1}{1-x}$ | $2^2, 2^1, 2^1, 2, 2, 2, ...$ |
| 44 | $Imm2$ | ⋊ ◐ | $\frac{x^3+x^2+x+1}{(x-1)^2}$ | $3^3, 6^3, 10^1, 14, 18, 22, ...$ |
| 45 | $Iba2$ | | $-\frac{(x+1)^2}{x-1}$ | $3^3, 4^1, 4, 4, 4, 4, ...$ |
| 46 | $Ima2$ | | $\frac{x^3+2x^2+2x+1}{1-x}$ | $3^3, 5^2, 6, 6, 6, 6, ...$ |
| 47 | $Pmmm$ | ⋊ ◐ | $-\frac{(x+1)^3}{(x-1)^3}$ | $6^6, 18, 38, 66, 102, 146, ...$ |
| 48 | $Pnnn$ | | $\frac{(x+1)^2}{(x-1)^2}$ | $4^4, 8, 12, 16, 20, 24, ...$ |
| 49 | $Pccm$ | | $\frac{(x+1)^3}{(x-1)^2}$ | $5^5, 12, 20, 28, 36, 44, ...$ |
| 50 | $Pban$ | | $\frac{(x+1)^2}{(x-1)^2}$ | $4^4, 8^1, 12, 16, 20, 24, ...$ |
| 51 | $Pmma$ | | $\frac{(x+1)^3}{(x-1)^2}$ | $5^5, 12, 20, 28, 36, 44, ...$ |
| 52 | $Pnna$ | | $\frac{x^3+2x^2+2x+1}{1-x}$ | $3^3, 5^2, 6, 6, 6, 6, ...$ |
| 53 | $Pmna$ | | $\frac{(x+1)^2}{(x-1)^2}$ | $4^4, 8^1, 12, 16, 20, 24, ...$ |
| 54 | $Pcca$ | | $-\frac{(x+1)^3}{x-1}$ | $4^4, 7, 8, 8, 8, 8, ...$ |
| 55 | $Pbam$ | | $\frac{(x+1)^2}{(x-1)^2}$ | $4^4, 8^1, 12, 16, 20, 24, ...$ |
| 56 | $Pccn$ | | $\frac{x^3+2x^2+2x+1}{1-x}$ | $3^3, 5^2, 6, 6, 6, 6, ...$ |
| 57 | $Pbcm$ | | $-\frac{(x+1)^3}{x-1}$ | $4^4, 7, 8, 8, 8, 8, ...$ |
| 58 | $Pnnm$ | | $\frac{x^3+x^2+x+1}{(x-1)^2}$ | $3^3, 6^3, 10^1, 14, 18, 22, ...$ |
| 59 | $Pmmn$ | | $\frac{(x+1)^2}{(x-1)^2}$ | $4^4, 8^1, 12, 16, 20, 24, ...$ |
| 60 | $Pbcn$ | | $-\frac{(x+1)^2}{x-1}$ | $3^3, 4^1, 4, 4, 4, 4, ...$ |
| 61 | $Pbca$ | | $\frac{x^3-2x-1}{x-1}$ | $3^3, 3, 2^1, 2, 2, 2, ...$ |
| 62 | $Pnma$ | | $\frac{x^3+2x^2+2x+1}{1-x}$ | $3^3, 5^2, 6, 6, 6, 6, ...$ |
| 63 | $Cmcm$ | | $\frac{(x+1)^2}{(x-1)^2}$ | $4^4, 8^1, 12, 16, 20, 24, ...$ |
| 64 | $Cmce$ | | $\frac{-x^3+2x+1}{(x-1)^2}$ | $4^4, 7, 9^1, 11, 13, 15, ...$ |
| 65 | $Cmmm$ | ⋊ ◐ | $-\frac{(x+1)^2}{(x-1)^3}$ | $5^5, 13^1, 25, 41, 61, 85, ...$ |
| 66 | $Cccm$ | | $\frac{x^3+2x^2+2x+1}{(x-1)^2}$ | $4^4, 9^2, 15, 21, 27, 33, ...$ |
| 67 | $Cmme$ | | $\frac{(x+1)^3}{(x-1)^2}$ | $5^5, 12, 20, 28, 36, 44, ...$ |
| 68 | $Ccce$ | | $\frac{-x^3+2x+1}{(x-1)^2}$ | $4^4, 7, 9^1, 11, 13, 15, ...$ |

| No. | Symbol | Marks | Polynomial | Sequence |
|---|---|---|---|---|
| 134 | $P4_2/nnm$ | | $\frac{x^3+2x^2+2x+1}{(x-1)^2}$ | $4^4, 9^1, 15, 21, 27, 33, ...$ |
| 135 | $P4_2/mbc$ | | $\frac{x^2+x+1}{(x-1)^2}$ | $3^3, 6^3, 9^1, 12, 15, 18, ...$ |
| 136 | $P4_2/mnm$ | | $-\frac{x^2+1}{(x-1)^3}$ | $3^3, 7^4, 13^2, 21^1, 31, 43, ...$ |
| 137 | $P4_2/nmc$ | | $\frac{x^2+x+1}{(x-1)^2}$ | $3^3, 6^3, 9^1, 12, 15, 18, ...$ |
| 138 | $P4_2/ncm$ | | $\frac{2x^2+x+1}{(x-1)^2}$ | $3^3, 7^4, 11, 15, 19, 23, ...$ |
| 139 | $I4/mmm$ | ⋊ | $-\frac{x^2+x+1}{(x-1)^3}$ | $4^4, 10^3, 19^1, 31, 46, 64, ...$ |
| 140 | $I4/mcm$ | | $\frac{x^3-x-1}{(x-1)^3}$ | $4^4, 9^2, 15, 22^1, 30, 39, ...$ |
| 141 | $I4_1/amd$ | | $\frac{x^3+x^2+x+1}{(x-1)^2}$ | $3^3, 6^2, 10^1, 14, 18, 22, ...$ |
| 142 | $I4_1/acd$ | | $\frac{-x^3+x+1}{(x-1)^2}$ | $3^3, 5^1, 6, 7^1, 8, 9, ...$ |
| 143 | $P3$ | ⋊ ◐ | $x^3+x^2+x+1$ | $1^1, 1^1, 1, 0, 0, 0, ...$ |
| 144 | $P3_1$ | ◐ ♭ | $x^3+x^2+x+1$ | $1^1, 1^1, 1, 0, 0, 0, ...$ |
| 145 | $P3_2$ | ◐ ♭ | $x^3+x^2+x+1$ | $1^1, 1^1, 1, 0, 0, 0, ...$ |
| 146 | $R3$ | ⋊ ◐ | $x^3+x^2+x+1$ | $1^1, 1^1, 1, 0, 0, 0, ...$ |
| 147 | $P\bar{3}$ | ⋊ ◐ | $\frac{x^3+x^2+x+1}{1-x}$ | $2^2, 3^1, 4, 4, 4, 4, ...$ |
| 148 | $R\bar{3}$ | ⋊ ◐ | $\frac{x^3+x^2+x+1}{1-x}$ | $2^2, 3^1, 4, 4, 4, 4, ...$ |
| 149 | $P312$ | ⋊ ◐ | $\frac{x^3+x^2+x+1}{1-x}$ | $2^2, 3^1, 4, 4, 4, 4, ...$ |
| 150 | $P321$ | ⋊ ◐ | $\frac{x^3+x^2+x+1}{1-x}$ | $2^2, 3^1, 4, 4, 4, 4, ...$ |
| 151 | $P3_112$ | ◐ | $\frac{x^3+x^2+x+1}{1-x}$ | $2^2, 3^1, 4, 4, 4, 4, ...$ |
| 152 | $P3_121$ | ◐ | $\frac{x^3+x^2+x+1}{1-x}$ | $2^2, 3^1, 4, 4, 4, 4, ...$ |
| 153 | $P3_212$ | ◐ | $\frac{x^3+x^2+x+1}{1-x}$ | $2^2, 3^1, 4, 4, 4, 4, ...$ |
| 154 | $P3_221$ | ◐ | $\frac{x^3+x^2+x+1}{1-x}$ | $2^2, 3^1, 4, 4, 4, 4, ...$ |
| 155 | $R32$ | ⋊ ◐ | $\frac{x^3+x^2+x+1}{1-x}$ | $2^2, 3^1, 4, 4, 4, 4, ...$ |
| 156 | $P3m1$ | ⋊ ◐ | $\frac{x^3+x^2+x+1}{1-x}$ | $2^2, 3^1, 4, 4, 4, 4, ...$ |
| 157 | $P31m$ | ⋊ ◐ | $\frac{x^3+x^2+x+1}{1-x}$ | $2^2, 3^1, 4, 4, 4, 4, ...$ |
| 158 | $P3c1$ | | $x^3+x^2+x+1$ | $1^1, 1^1, 1, 0, 0, 0, ...$ |
| 159 | $P31c$ | | $x^3+x^2+x+1$ | $1^1, 1^1, 1, 0, 0, 0, ...$ |
| 160 | $R3m$ | ⋊ ◐ | $\frac{x^3+x^2+x+1}{1-x}$ | $2^2, 3^1, 4, 4, 4, 4, ...$ |
| 161 | $R3c$ | | $x^3+x^2+x+1$ | $1^1, 1^1, 1, 0, 0, 0, ...$ |
| 162 | $P\bar{3}1m$ | ⋊ ◐ | $\frac{x^3+x^2+x+1}{(x-1)^2}$ | $3^3, 6^1, 10, 14, 18, 22, ...$ |
| 163 | $P\bar{3}1c$ | | $\frac{x^3+x^2+x+1}{1-x}$ | $2^2, 3^1, 4, 4, 4, 4, ...$ |
| 164 | $P\bar{3}m1$ | ⋊ ◐ | $\frac{x^3+x^2+x+1}{(x-1)^2}$ | $3^3, 6^1, 10, 14, 18, 22, ...$ |
| 165 | $P\bar{3}c1$ | | $\frac{x^3+x^2+x+1}{1-x}$ | $2^2, 3^1, 4, 4, 4, 4, ...$ |
| 166 | $R\bar{3}m$ | ⋊ ◐ | $\frac{x^3+x^2+x+1}{(x-1)^2}$ | $3^3, 6^1, 10, 14, 18, 22, ...$ |
| 167 | $R\bar{3}c$ | | $\frac{x^3+x^2+x+1}{1-x}$ | $2^2, 3^1, 4, 4, 4, 4, ...$ |
| 168 | $P6$ | ⋊ ◐ | $\frac{x^3+x^2+x+1}{1-x}$ | $2^2, 3^1, 4, 4, 4, 4, ...$ |
| 169 | $P6_1$ | ♭ | $x^3+x^2+x+1$ | $1^1, 1^1, 1, 0, 0, 0, ...$ |
| 170 | $P6_5$ | ♭ | $x^3+x^2+x+1$ | $1^1, 1^1, 1, 0, 0, 0, ...$ |
| 171 | $P6_2$ | ◐ | $\frac{x^3+x^2+x+1}{1-x}$ | $2^2, 3^1, 4, 4, 4, 4, ...$ |
| 172 | $P6_4$ | ◐ | $\frac{x^3+x^2+x+1}{1-x}$ | $2^2, 3^1, 4, 4, 4, 4, ...$ |
| 173 | $P6_3$ | | $x^3+x^2+x+1$ | $1^1, 1^1, 1, 0, 0, 0, ...$ |
| 174 | $P\bar{6}$ | ⋊ ◐ | $\frac{x^3+x^2+x+1}{1-x}$ | $2^2, 3^1, 4, 4, 4, 4, ...$ |
| 175 | $P6/m$ | ⋊ ◐ | $\frac{x^3+x^2+x+1}{(x-1)^2}$ | $3^3, 6^1, 10, 14, 18, 22, ...$ |
| 176 | $P6_3/m$ | | $\frac{x^3+x^2+x+1}{1-x}$ | $2^2, 3^1, 4, 4, 4, 4, ...$ |
| 177 | $P622$ | ⋊ ◐ | $\frac{x^3+x^2+x+1}{(x-1)^2}$ | $3^3, 6^1, 10, 14, 18, 22, ...$ |
| 178 | $P6_122$ | | $\frac{x^3+x^2+x+1}{1-x}$ | $2^2, 3^1, 4, 4, 4, 4, ...$ |
| 179 | $P6_522$ | | $\frac{x^3+x^2+x+1}{1-x}$ | $2^2, 3^1, 4, 4, 4, 4, ...$ |
| 180 | $P6_222$ | ◐ | $\frac{x^3+x^2+x+1}{(x-1)^2}$ | $3^3, 6^1, 10, 14, 18, 22, ...$ |
| 181 | $P6_422$ | ◐ | $\frac{x^3+x^2+x+1}{(x-1)^2}$ | $3^3, 6^1, 10, 14, 18, 22, ...$ |
| 182 | $P6_322$ | | $\frac{x^3+x^2+x+1}{1-x}$ | $2^2, 3^1, 4, 4, 4, 4, ...$ |
| 183 | $P6mm$ | ⋊ ◐ | $\frac{x^3+x^2+x+1}{(x-1)^2}$ | $3^3, 6^1, 10, 14, 18, 22, ...$ |

| No. | Group | | Function | Sequence |
|---|---|---|---|---|
| 69 | $Fmmm$ | ⋊ | $\frac{x^3-2x-1}{(x-1)^3}$ | $5^5, 12, 21^1, 32, 45, 60, ...$ |
| 70 | $Fddd$ | | $\frac{x+1}{(x-1)^2}$ | $3^3, 5^1, 7^1, 9, 11, 13, ...$ |
| 71 | $Immm$ | ⋊ ☾ | $-\frac{x^3+x^2+x+1}{(x-1)^3}$ | $4^4, 10^3, 20^1, 34, 52, 74, ...$ |
| 72 | $Ibam$ | | $\frac{(x+1)^2}{(x-1)^2}$ | $4^4, 8^1, 12, 16, 20, 24, ...$ |
| 73 | $Ibca$ | | $-\frac{(x+1)^3}{x-1}$ | $4^4, 7, 8, 8, 8, 8, ...$ |
| 74 | $Imma$ | | $\frac{x^3+2x^2+2x+1}{(x-1)^2}$ | $4^4, 9^2, 15, 21, 27, 33, ...$ |
| 75 | $P4$ | ⋊ ☾ | $\frac{x^3+2x^2+2x+1}{1-x}$ | $3^3, 5^2, 6, 6, 6, 6, ...$ |
| 76 | $P4_1$ | ♭ | $x^3+2x^2+2x+1$ | $2^2, 2^1, 1, 0, 0, 0, ...$ |
| 77 | $P4_2$ | ☾ | $\frac{x^3+2x^2+2x+1}{1-x}$ | $3^3, 5^1, 6, 6, 6, 6, ...$ |
| 78 | $P4_3$ | ♭ | $x^3+2x^2+2x+1$ | $2^2, 2^1, 1, 0, 0, 0, ...$ |
| 79 | $I4$ | ⋊ ☾ | $\frac{x^3+x^2+x+1}{1-x}$ | $2^2, 3^3, 4^2, 4, 4, 4, ...$ |
| 80 | $I4_1$ | | $\frac{x+1}{1-x}$ | $2^2, 2^1, 2^1, 2, 2, 2, ...$ |
| 81 | $P\bar{4}$ | ⋊ ☾ | $\frac{x^3+2x^2+2x+1}{1-x}$ | $3^3, 5^2, 6, 6, 6, 6, ...$ |
| 82 | $I\bar{4}$ | ⋊ ☾ | $\frac{x^3+x^2+x+1}{1-x}$ | $2^2, 3^3, 4^2, 4, 4, 4, ...$ |
| 83 | $P4/m$ | ⋊ ☾ | $\frac{x^3+2x^2+2x+1}{(x-1)^2}$ | $4^4, 9^2, 15, 21, 27, 33, ...$ |
| 84 | $P4_2/m$ | | $\frac{2x^2+1}{(x-1)^2}$ | $3^3, 7^4, 11, 15, 19, 23, ...$ |
| 85 | $P4/n$ | | $\frac{x^3+2x^2+2x+1}{1-x}$ | $3^3, 5^2, 6, 6, 6, 6, ...$ |
| 86 | $P4_2/n$ | | $\frac{x^3+2x^2+2x+1}{1-x}$ | $3^3, 5^1, 6, 6, 6, 6, ...$ |
| 87 | $I4/m$ | ⋊ | $\frac{x^2+x+1}{(x-1)^2}$ | $3^3, 6^3, 9^1, 12, 15, 18, ...$ |
| 88 | $I4_1/a$ | | $\frac{x^3+x^2+x+1}{1-x}$ | $2^2, 3^2, 4^1, 4, 4, 4, ...$ |
| 89 | $P422$ | ⋊ ☾ | $\frac{x^3+2x^2+2x+1}{(x-1)^2}$ | $4^4, 9^2, 15, 21, 27, 33, ...$ |
| 90 | $P42_12$ | | $\frac{x+1}{(x-1)^2}$ | $3^3, 5^2, 7^1, 9, 11, 13, ...$ |
| 91 | $P4_122$ | | $\frac{x^3+2x^2+2x+1}{1-x}$ | $3^3, 5^1, 6, 6, 6, 6, ...$ |
| 92 | $P4_12_12$ | | $\frac{x+1}{1-x}$ | $2^2, 2^1, 2^1, 2, 2, 2, ...$ |
| 93 | $P4_222$ | ☾ | $\frac{x^3+2x^2+2x+1}{(x-1)^2}$ | $4^4, 9^1, 15, 21, 27, 33, ...$ |
| 94 | $P4_22_12$ | | $\frac{x+1}{(x-1)^2}$ | $3^3, 5^1, 7^1, 9, 11, 13, ...$ |
| 95 | $P4_322$ | | $\frac{x^3+2x^2+2x+1}{1-x}$ | $3^3, 5^1, 6, 6, 6, 6, ...$ |
| 96 | $P4_32_12$ | | $\frac{x+1}{1-x}$ | $2^2, 2^1, 2^1, 2, 2, 2, ...$ |
| 97 | $I422$ | ⋊ ☾ | $\frac{x^3+x^2+x+1}{(x-1)^2}$ | $3^3, 6^3, 10^2, 14, 18, 22, ...$ |
| 98 | $I4_122$ | | $\frac{x+1}{(x-1)^2}$ | $3^3, 5^1, 7^1, 9, 11, 13, ...$ |
| 99 | $P4mm$ | ⋊ ☾ | $\frac{x^3+2x^2+2x+1}{(x-1)^2}$ | $4^4, 9^2, 15, 21, 27, 33, ...$ |
| 100 | $P4bm$ | | $\frac{x+1}{(x-1)^2}$ | $3^3, 5^2, 7^1, 9, 11, 13, ...$ |
| 101 | $P4_2cm$ | | $\frac{-x^3+x^2+x+1}{(x-1)^2}$ | $3^3, 6^3, 8, 10, 12, 14, ...$ |
| 102 | $P4_2nm$ | | $\frac{x+1}{(x-1)^2}$ | $3^3, 5^1, 7^1, 9, 11, 13, ...$ |
| 103 | $P4cc$ | | $\frac{x^3+2x^2+2x+1}{1-x}$ | $3^3, 5^2, 6, 6, 6, 6, ...$ |
| 104 | $P4nc$ | | $\frac{x^3+x^2+x+1}{1-x}$ | $2^2, 3^3, 4^2, 4, 4, 4, ...$ |
| 105 | $P4_2mc$ | | $\frac{2x^2+x+1}{(x-1)^2}$ | $3^3, 7^4, 11, 15, 19, 23, ...$ |
| 106 | $P4_2bc$ | | $\frac{x^3+x^2+x+1}{1-x}$ | $2^2, 3^3, 4^2, 4, 4, 4, ...$ |
| 107 | $I4mm$ | ⋊ | $\frac{x^2+x+1}{(x-1)^2}$ | $3^3, 6^3, 9^1, 12, 15, 18, ...$ |
| 108 | $I4cm$ | | $\frac{-x^3+x+1}{(x-1)^2}$ | $3^3, 5^2, 6, 7^1, 8, 9, ...$ |
| 109 | $I4_1md$ | | $\frac{x^3+1}{(x-1)^2}$ | $2^2, 3^2, 5^2, 7^1, 9, 11, ...$ |
| 110 | $I4_1cd$ | | $\frac{x+1}{1-x}$ | $2^2, 2^1, 2^1, 2, 2, 2, ...$ |
| 111 | $P\bar{4}2m$ | ⋊ ☾ | $\frac{x^3+2x^2+2x+1}{(x-1)^2}$ | $4^4, 9^2, 15, 21, 27, 33, ...$ |
| 112 | $P\bar{4}2c$ | | $\frac{2x^2+x+1}{(x-1)^2}$ | $3^3, 7^4, 11, 15, 19, 23, ...$ |
| 113 | $P\bar{4}2_1m$ | | $\frac{x+1}{(x-1)^2}$ | $3^3, 5^2, 7^1, 9, 11, 13, ...$ |

| No. | Group | | Function | Sequence |
|---|---|---|---|---|
| 184 | $P6cc$ | | $\frac{x^3+x^2+x+1}{1-x}$ | $2^2, 3^1, 4, 4, 4, 4, ...$ |
| 185 | $P6_3cm$ | | $\frac{x^3+x^2+x+1}{1-x}$ | $2^2, 3^1, 4, 4, 4, 4, ...$ |
| 186 | $P6_3mc$ | | $\frac{x^3+x^2+x+1}{1-x}$ | $2^2, 3^1, 4, 4, 4, 4, ...$ |
| 187 | $P\bar{6}m2$ | ⋊ ☾ | $\frac{x^3+x^2+x+1}{(x-1)^2}$ | $3^3, 6^1, 10, 14, 18, 22, ...$ |
| 188 | $P\bar{6}c2$ | | $\frac{x^3+x^2+x+1}{1-x}$ | $2^2, 3^1, 4, 4, 4, 4, ...$ |
| 189 | $P\bar{6}2m$ | ⋊ ☾ | $\frac{x^3+x^2+x+1}{(x-1)^2}$ | $3^3, 6^1, 10, 14, 18, 22, ...$ |
| 190 | $P\bar{6}2c$ | | $\frac{x^3+x^2+x+1}{1-x}$ | $2^2, 3^1, 4, 4, 4, 4, ...$ |
| 191 | $P6/mmm$ | ⋊ ☾ | $-\frac{x^3+x^2+x+1}{(x-1)^3}$ | $4^4, 10^1, 20, 34, 52, 74, ...$ |
| 192 | $P6/mcc$ | | $\frac{x^3+x^2+x+1}{(x-1)^2}$ | $3^3, 6^1, 10, 14, 18, 22, ...$ |
| 193 | $P6_3/mcm$ | | $\frac{x^3+x^2+x+1}{(x-1)^2}$ | $3^3, 6^1, 10, 14, 18, 22, ...$ |
| 194 | $P6_3/mmc$ | | $\frac{x^3+x^2+x+1}{(x-1)^2}$ | $3^3, 6^1, 10, 14, 18, 22, ...$ |
| 195 | $P23$ | ⋊ ☾ | $\frac{x^5+3x^3+3x^2+1}{(x-1)^2(x^2+x+1)}$ | $1^1, 4^3, 8^4, 8, 12, 16, ...$ |
| 196 | $F23$ | ⋊ ☾ | $\frac{x^6+x^5-3x^4+2x^3+3x^2-x+1}{(x-1)^2(x^2+x+1)}$ | $0, 3^3, 6^6, 2, 6, 10, ...$ |
| 197 | $I23$ | ⋊ ☾ | $\frac{x^5+x^3+x^2+1}{(x-1)^2(x^2+x+1)}$ | $1^1, 2^2, 4^4, 4^1, 6, 8, ...$ |
| 198 | $P2_13$ | | $x^3+1$ | $0, 0, 1^1, 0, 0, 0, ...$ |
| 199 | $I2_13$ | | $\frac{x^3+1}{1-x}$ | $1^1, 1, 2^1, 2, 2, 2, ...$ |
| 200 | $Pm\bar{3}$ | ⋊ ☾ | $-\frac{x^5+3x^3+3x^2+1}{(x-1)^3(x^2+x+1)}$ | $2^2, 6^3, 14^4, 22, 34, 50, ...$ |
| 201 | $Pn\bar{3}$ | | $\frac{-x^4+x^3+2x^2+1}{(x-1)^2(x^2+x+1)}$ | $2^2, 4^1, 6^2, 6, 8, 10, ...$ |
| 202 | $Fm\bar{3}$ | ⋊ | $-\frac{x^5-3x^4+x^3+3x^2-x+1}{(x-1)^3(x^2+x+1)}$ | $1^1, 4^3, 9^5, 10, 15, 22, ...$ |
| 203 | $Fd\bar{3}$ | | $\frac{-2x^4+x^3+2x^2+1}{(x-1)^2(x^2+x+1)}$ | $1^1, 3^2, 5^3, 3, 5, 7, ...$ |
| 204 | $Im\bar{3}$ | ⋊ ☾ | $-\frac{x^5+x^3+x^2+1}{(x-1)^3(x^2+x+1)}$ | $2^2, 4^2, 8^4, 12^1, 18, 26, ...$ |
| 205 | $Pa\bar{3}$ | | $\frac{x^3+1}{1-x}$ | $1^1, 1, 2^1, 2, 2, 2, ...$ |
| 206 | $Ia\bar{3}$ | | $\frac{x^3+x^2+x+1}{1-x}$ | $2^2, 3, 4, 4, 4, 4, ...$ |
| 207 | $P432$ | ⋊ ☾ | $\frac{x^5+x^4+3x^3+3x^2+x+1}{(x-1)^2(x^2+x+1)}$ | $2^2, 5^2, 9^2, 11, 15, 19, ...$ |
| 208 | $P4_232$ | ☾ | $\frac{x^5+x^4+3x^3+3x^2+x+1}{(x-1)^2(x^2+x+1)}$ | $2^2, 5^2, 9^2, 11, 15, 19, ...$ |
| 209 | $F432$ | ⋊ ☾ | $\frac{x^5+3x^3+3x^2+1}{(x-1)^2(x^2+x+1)}$ | $1^1, 4^3, 8^4, 8, 12, 16, ...$ |
| 210 | $F4_132$ | | $\frac{-2x^4+x^3+2x^2+1}{(x-1)^2(x^2+x+1)}$ | $1^1, 3^2, 5^3, 3, 5, 7, ...$ |
| 211 | $I432$ | ⋊ ☾ | $\frac{(x+1)(x^2+1)^2}{(x-1)^2(x^2+x+1)}$ | $2^2, 4^2, 7^3, 9, 12, 15, ...$ |
| 212 | $P4_332$ | | $\frac{x^3+1}{1-x}$ | $1^1, 1, 2^1, 2, 2, 2, ...$ |
| 213 | $P4_132$ | | $\frac{x^3+1}{1-x}$ | $1^1, 1, 2^1, 2, 2, 2, ...$ |
| 214 | $I4_132$ | | $\frac{x^3+1}{(x-1)^2}$ | $2^2, 3, 5^1, 7, 9, 11, ...$ |
| 215 | $P\bar{4}3m$ | ⋊ ☾ | $\frac{x^5+x^4+3x^3+3x^2+x+1}{(x-1)^2(x^2+x+1)}$ | $2^2, 5^2, 9^2, 11, 15, 19, ...$ |
| 216 | $F\bar{4}3m$ | ⋊ ☾ | $\frac{x^5+3x^3+3x^2+1}{(x-1)^2(x^2+x+1)}$ | $1^1, 4^3, 8^4, 8, 12, 16, ...$ |
| 217 | $I\bar{4}3m$ | ⋊ | $\frac{x^3+2x^2+x+1}{(x-1)^2(x^2+x+1)}$ | $2^2, 4^2, 6^2, 7, 9, 11, ...$ |
| 218 | $P\bar{4}3n$ | | $\frac{x^3+2x^2+1}{(x-1)^2(x^2+x+1)}$ | $1^1, 3^3, 5^3, 5, 7, 9, ...$ |
| 219 | $F\bar{4}3c$ | | $\frac{x^6-2x^4+2x^2+1}{(x-1)^2(x^2+x+1)}$ | $1^1, 3^2, 4^2, 2, 4, 6^2, ...$ |
| 220 | $I\bar{4}3d$ | | $\frac{x^3+1}{1-x}$ | $1^1, 1^1, 2^1, 2, 2, 2, ...$ |
| 221 | $Pm\bar{3}m$ | ⋊ ☾ | $-\frac{x^5+x^4+3x^3+3x^2+x+1}{(x-1)^3(x^2+x+1)}$ | $3^3, 8^2, 17^2, 28, 43, 62, ...$ |
| 222 | $Pn\bar{3}n$ | | $\frac{x^3+2x^2+x+1}{(x-1)^2(x^2+x+1)}$ | $2^2, 4^2, 6^2, 7, 9, 11, ...$ |
| 223 | $Pm\bar{3}n$ | | $-\frac{x^3+2x^2+1}{(x-1)^3(x^2+x+1)}$ | $2^2, 5^3, 10^3, 15, 22, 31, ...$ |
| 224 | $Pn\bar{3}m$ | | $\frac{x^5+2x^4+4x^3+4x^2+2x+1}{(x-1)^2(x^2+x+1)}$ | $3^3, 7^1, 12^1, 16, 21, 26, ...$ |
| 225 | $Fm\bar{3}m$ | ⋊ | $-\frac{x^5+2x^3+3x^2+1}{(x-1)^3(x^2+x+1)}$ | $2^2, 6^3, 13^3, 20, 31, 45, ...$ |
| 226 | $Fm\bar{3}c$ | | $\frac{2x^4-2x^2-1}{(x-1)^3(x^2+x+1)}$ | $2^2, 5^2, 9^2, 11, 15, 20^2, ...$ |
| 227 | $Fd\bar{3}m$ | | $\frac{x^5+x^4+3x^3+3x^2+x+1}{(x-1)^2(x^2+x+1)}$ | $2^2, 5^2, 9^2, 11, 15, 19, ...$ |
| 228 | $Fd\bar{3}c$ | | $\frac{-x^5-2x^4+2x^2+x+1}{(x-1)^2(x^2+x+1)}$ | $2^2, 4^1, 5^1, 4, 5, 6^1, ...$ |

| 114 | $P\bar{4}2_1c$ | | $\frac{x^3+x^2+x+1}{1-x}$ | $2^2, 3^3, 4^2, 4, 4, 4, ...$ | 229 | $Im\bar{3}m$ | $\rtimes$ | $-\frac{x^3+2x^2+x+1}{(x-1)^3(x^2+x+1)}$ | $3^3, 7^2, 13^2, 20, 29, 40, ...$ |
|---|---|---|---|---|---|---|---|---|---|
| 115 | $P\bar{4}m2$ | $\rtimes$ ◐ | $\frac{x^3+2x^2+2x+1}{(x-1)^2}$ | $4^4, 9^2, 15, 21, 27, 33, ...$ | 230 | $Ia\bar{3}d$ | | $\frac{1}{(x-1)^2}$ | $2^2, 3^1, 4, 5^1, 6, 7, ...$ |

TABLE V: Integral cohomology groups for the 230 space groups $G$ with trivial action ($H^n(G,\mathbb{Z})$) or the orientation-reversing action ($H^n(G,\mathbb{Z}^{\mathrm{or}})$) up to $n=6$.

| No. | $H^n(G,\mathbb{Z})$ | | | | | | $H^n(G,\mathbb{Z}^{\mathrm{or}})$ | | | | | |
|---|---|---|---|---|---|---|---|---|---|---|---|---|
| | $n=1$ | $n=2$ | $n=3$ | $n=4$ | $n=5$ | $n=6$ | $n=1$ | $n=2$ | $n=3$ | $n=4$ | $n=5$ | $n=6$ |
| 1 | $\mathbb{Z}^3$ | $\mathbb{Z}^3$ | $\mathbb{Z}$ | $\mathbb{Z}_1$ | $\mathbb{Z}_1$ | $\mathbb{Z}_1$ | $\mathbb{Z}^3$ | $\mathbb{Z}^3$ | $\mathbb{Z}$ | $\mathbb{Z}_1$ | $\mathbb{Z}_1$ | $\mathbb{Z}_1$ |
| 2 | $\mathbb{Z}_1$ | $\mathbb{Z}^3\oplus\mathbb{Z}_2^4$ | $\mathbb{Z}_1$ | $\mathbb{Z}_2^8$ | $\mathbb{Z}_1$ | $\mathbb{Z}_2^8$ | $\mathbb{Z}^3\oplus\mathbb{Z}_2$ | $\mathbb{Z}_1$ | $\mathbb{Z}\oplus\mathbb{Z}_2^7$ | $\mathbb{Z}_1$ | $\mathbb{Z}_2^8$ | $\mathbb{Z}_1$ |
| 3 | $\mathbb{Z}$ | $\mathbb{Z}\oplus\mathbb{Z}_2^3$ | $\mathbb{Z}\oplus\mathbb{Z}_2^3$ | $\mathbb{Z}_2^4$ | $\mathbb{Z}_2^4$ | $\mathbb{Z}_2^4$ | $\mathbb{Z}$ | $\mathbb{Z}\oplus\mathbb{Z}_2^3$ | $\mathbb{Z}\oplus\mathbb{Z}_2^3$ | $\mathbb{Z}_2^4$ | $\mathbb{Z}_2^4$ | $\mathbb{Z}_2^4$ |
| 4 | $\mathbb{Z}$ | $\mathbb{Z}\oplus\mathbb{Z}_2^2$ | $\mathbb{Z}$ | $\mathbb{Z}_1$ | $\mathbb{Z}_1$ | $\mathbb{Z}_1$ | $\mathbb{Z}$ | $\mathbb{Z}\oplus\mathbb{Z}_2^2$ | $\mathbb{Z}$ | $\mathbb{Z}_1$ | $\mathbb{Z}_1$ | $\mathbb{Z}_1$ |
| 5 | $\mathbb{Z}$ | $\mathbb{Z}\oplus\mathbb{Z}_2^2$ | $\mathbb{Z}\oplus\mathbb{Z}_2$ | $\mathbb{Z}_2^2$ | $\mathbb{Z}_2^2$ | $\mathbb{Z}_2^2$ | $\mathbb{Z}$ | $\mathbb{Z}\oplus\mathbb{Z}_2^2$ | $\mathbb{Z}\oplus\mathbb{Z}_2$ | $\mathbb{Z}_2^2$ | $\mathbb{Z}_2^2$ | $\mathbb{Z}_2^2$ |
| 6 | $\mathbb{Z}^2$ | $\mathbb{Z}\oplus\mathbb{Z}_2^2$ | $\mathbb{Z}_2^4$ | $\mathbb{Z}_2^4$ | $\mathbb{Z}_2^4$ | $\mathbb{Z}_2^4$ | $\mathbb{Z}\oplus\mathbb{Z}_2$ | $\mathbb{Z}^2\oplus\mathbb{Z}_2^2$ | $\mathbb{Z}\oplus\mathbb{Z}_2^3$ | $\mathbb{Z}_2^4$ | $\mathbb{Z}_2^4$ | $\mathbb{Z}_2^4$ |
| 7 | $\mathbb{Z}^2$ | $\mathbb{Z}\oplus\mathbb{Z}_2$ | $\mathbb{Z}_2$ | $\mathbb{Z}_1$ | $\mathbb{Z}_1$ | $\mathbb{Z}_1$ | $\mathbb{Z}\oplus\mathbb{Z}_2$ | $\mathbb{Z}^2\oplus\mathbb{Z}_2$ | $\mathbb{Z}$ | $\mathbb{Z}_1$ | $\mathbb{Z}_1$ | $\mathbb{Z}_1$ |
| 8 | $\mathbb{Z}^2$ | $\mathbb{Z}\oplus\mathbb{Z}_2$ | $\mathbb{Z}_2^2$ | $\mathbb{Z}_2^2$ | $\mathbb{Z}_2^2$ | $\mathbb{Z}_2^2$ | $\mathbb{Z}\oplus\mathbb{Z}_2$ | $\mathbb{Z}^2\oplus\mathbb{Z}_2$ | $\mathbb{Z}\oplus\mathbb{Z}_2$ | $\mathbb{Z}_2^2$ | $\mathbb{Z}_2^2$ | $\mathbb{Z}_2^2$ |
| 9 | $\mathbb{Z}^2$ | $\mathbb{Z}$ | $\mathbb{Z}_2$ | $\mathbb{Z}_1$ | $\mathbb{Z}_1$ | $\mathbb{Z}_1$ | $\mathbb{Z}\oplus\mathbb{Z}_2$ | $\mathbb{Z}^2$ | $\mathbb{Z}$ | $\mathbb{Z}_1$ | $\mathbb{Z}_1$ | $\mathbb{Z}_1$ |
| 10 | $\mathbb{Z}_1$ | $\mathbb{Z}\oplus\mathbb{Z}_2^5$ | $\mathbb{Z}_2^6$ | $\mathbb{Z}_2^{14}$ | $\mathbb{Z}_2^{14}$ | $\mathbb{Z}_2^{22}$ | $\mathbb{Z}\oplus\mathbb{Z}_2$ | $\mathbb{Z}_2^3$ | $\mathbb{Z}\oplus\mathbb{Z}_2^9$ | $\mathbb{Z}_2^{10}$ | $\mathbb{Z}_2^{18}$ | $\mathbb{Z}_2^{18}$ |
| 11 | $\mathbb{Z}_1$ | $\mathbb{Z}\oplus\mathbb{Z}_2^4$ | $\mathbb{Z}_2^2$ | $\mathbb{Z}_2^6$ | $\mathbb{Z}_2^2$ | $\mathbb{Z}_2^6$ | $\mathbb{Z}\oplus\mathbb{Z}_2$ | $\mathbb{Z}_2^2$ | $\mathbb{Z}\oplus\mathbb{Z}_2^5$ | $\mathbb{Z}_2^2$ | $\mathbb{Z}_2^6$ | $\mathbb{Z}_2^2$ |
| 12 | $\mathbb{Z}_1$ | $\mathbb{Z}\oplus\mathbb{Z}_2^4$ | $\mathbb{Z}_2^3$ | $\mathbb{Z}_2^9$ | $\mathbb{Z}_2^7$ | $\mathbb{Z}_2^{13}$ | $\mathbb{Z}\oplus\mathbb{Z}_2$ | $\mathbb{Z}_2^2$ | $\mathbb{Z}\oplus\mathbb{Z}_2^6$ | $\mathbb{Z}_2^5$ | $\mathbb{Z}_2^{11}$ | $\mathbb{Z}_2^9$ |
| 13 | $\mathbb{Z}_1$ | $\mathbb{Z}\oplus\mathbb{Z}_2^4$ | $\mathbb{Z}_2^2$ | $\mathbb{Z}_2^6$ | $\mathbb{Z}_2^2$ | $\mathbb{Z}_2^6$ | $\mathbb{Z}\oplus\mathbb{Z}_2$ | $\mathbb{Z}_2^2$ | $\mathbb{Z}\oplus\mathbb{Z}_2^5$ | $\mathbb{Z}_2^2$ | $\mathbb{Z}_2^6$ | $\mathbb{Z}_2^2$ |
| 14 | $\mathbb{Z}_1$ | $\mathbb{Z}\oplus\mathbb{Z}_2^2\oplus\mathbb{Z}_4$ | $\mathbb{Z}_1$ | $\mathbb{Z}_2^4$ | $\mathbb{Z}_1$ | $\mathbb{Z}_2^4$ | $\mathbb{Z}\oplus\mathbb{Z}_2$ | $\mathbb{Z}_2$ | $\mathbb{Z}\oplus\mathbb{Z}_2^3$ | $\mathbb{Z}_1$ | $\mathbb{Z}_2^4$ | $\mathbb{Z}_1$ |
| 15 | $\mathbb{Z}_1$ | $\mathbb{Z}\oplus\mathbb{Z}_2^3$ | $\mathbb{Z}_2$ | $\mathbb{Z}_2^5$ | $\mathbb{Z}_2$ | $\mathbb{Z}_2^5$ | $\mathbb{Z}\oplus\mathbb{Z}_2$ | $\mathbb{Z}_2$ | $\mathbb{Z}\oplus\mathbb{Z}_2^4$ | $\mathbb{Z}_2$ | $\mathbb{Z}_2^5$ | $\mathbb{Z}_2$ |
| 16 | $\mathbb{Z}_1$ | $\mathbb{Z}_2^5$ | $\mathbb{Z}\oplus\mathbb{Z}_2^7$ | $\mathbb{Z}_2^{12}$ | $\mathbb{Z}_2^{16}$ | $\mathbb{Z}_2^{20}$ | $\mathbb{Z}_1$ | $\mathbb{Z}_2^5$ | $\mathbb{Z}\oplus\mathbb{Z}_2^7$ | $\mathbb{Z}_2^{12}$ | $\mathbb{Z}_2^{16}$ | $\mathbb{Z}_2^{20}$ |
| 17 | $\mathbb{Z}_1$ | $\mathbb{Z}_2^4$ | $\mathbb{Z}\oplus\mathbb{Z}_2^3$ | $\mathbb{Z}_2^4$ | $\mathbb{Z}_2^4$ | $\mathbb{Z}_2^4$ | $\mathbb{Z}_1$ | $\mathbb{Z}_2^4$ | $\mathbb{Z}\oplus\mathbb{Z}_2^3$ | $\mathbb{Z}_2^4$ | $\mathbb{Z}_2^4$ | $\mathbb{Z}_2^4$ |
| 18 | $\mathbb{Z}_1$ | $\mathbb{Z}_2^2\oplus\mathbb{Z}_4$ | $\mathbb{Z}\oplus\mathbb{Z}_2$ | $\mathbb{Z}_2^2$ | $\mathbb{Z}_2^2$ | $\mathbb{Z}_2^2$ | $\mathbb{Z}_1$ | $\mathbb{Z}_2^2\oplus\mathbb{Z}_4$ | $\mathbb{Z}\oplus\mathbb{Z}_2$ | $\mathbb{Z}_2^2$ | $\mathbb{Z}_2^2$ | $\mathbb{Z}_2^2$ |
| 19 | $\mathbb{Z}_1$ | $\mathbb{Z}_4^2$ | $\mathbb{Z}$ | $\mathbb{Z}_1$ | $\mathbb{Z}_1$ | $\mathbb{Z}_1$ | $\mathbb{Z}_1$ | $\mathbb{Z}_4^2$ | $\mathbb{Z}$ | $\mathbb{Z}_1$ | $\mathbb{Z}_1$ | $\mathbb{Z}_1$ |
| 20 | $\mathbb{Z}_1$ | $\mathbb{Z}_2^3$ | $\mathbb{Z}\oplus\mathbb{Z}_2$ | $\mathbb{Z}_2^2$ | $\mathbb{Z}_2^2$ | $\mathbb{Z}_2^2$ | $\mathbb{Z}_1$ | $\mathbb{Z}_2^3$ | $\mathbb{Z}\oplus\mathbb{Z}_2$ | $\mathbb{Z}_2^2$ | $\mathbb{Z}_2^2$ | $\mathbb{Z}_2^2$ |
| 21 | $\mathbb{Z}_1$ | $\mathbb{Z}_2^4$ | $\mathbb{Z}\oplus\mathbb{Z}_2^4$ | $\mathbb{Z}_2^7$ | $\mathbb{Z}_2^9$ | $\mathbb{Z}_2^{11}$ | $\mathbb{Z}_1$ | $\mathbb{Z}_2^4$ | $\mathbb{Z}\oplus\mathbb{Z}_2^4$ | $\mathbb{Z}_2^7$ | $\mathbb{Z}_2^9$ | $\mathbb{Z}_2^{11}$ |
| 22 | $\mathbb{Z}_1$ | $\mathbb{Z}_2^4$ | $\mathbb{Z}\oplus\mathbb{Z}_2^2\oplus\mathbb{Z}_4$ | $\mathbb{Z}_2^6$ | $\mathbb{Z}_2^8$ | $\mathbb{Z}_2^{10}$ | $\mathbb{Z}_1$ | $\mathbb{Z}_2^4$ | $\mathbb{Z}\oplus\mathbb{Z}_2^2\oplus\mathbb{Z}_4$ | $\mathbb{Z}_2^6$ | $\mathbb{Z}_2^8$ | $\mathbb{Z}_2^{10}$ |
| 23 | $\mathbb{Z}_1$ | $\mathbb{Z}_2^2\oplus\mathbb{Z}_4$ | $\mathbb{Z}\oplus\mathbb{Z}_2^3$ | $\mathbb{Z}_2^6$ | $\mathbb{Z}_2^8$ | $\mathbb{Z}_2^{10}$ | $\mathbb{Z}_1$ | $\mathbb{Z}_2^2\oplus\mathbb{Z}_4$ | $\mathbb{Z}\oplus\mathbb{Z}_2^3$ | $\mathbb{Z}_2^6$ | $\mathbb{Z}_2^8$ | $\mathbb{Z}_2^{10}$ |
| 24 | $\mathbb{Z}_1$ | $\mathbb{Z}_2^3$ | $\mathbb{Z}\oplus\mathbb{Z}_2^2$ | $\mathbb{Z}_2^3$ | $\mathbb{Z}_2^3$ | $\mathbb{Z}_2^3$ | $\mathbb{Z}_1$ | $\mathbb{Z}_2^3$ | $\mathbb{Z}\oplus\mathbb{Z}_2^2$ | $\mathbb{Z}_2^3$ | $\mathbb{Z}_2^3$ | $\mathbb{Z}_2^3$ |
| 25 | $\mathbb{Z}$ | $\mathbb{Z}_2^4$ | $\mathbb{Z}_2^8$ | $\mathbb{Z}_2^{12}$ | $\mathbb{Z}_2^{16}$ | $\mathbb{Z}_2^{20}$ | $\mathbb{Z}_2$ | $\mathbb{Z}\oplus\mathbb{Z}_2^4$ | $\mathbb{Z}\oplus\mathbb{Z}_2^7$ | $\mathbb{Z}_2^{12}$ | $\mathbb{Z}_2^{16}$ | $\mathbb{Z}_2^{20}$ |
| 26 | $\mathbb{Z}$ | $\mathbb{Z}_2^3$ | $\mathbb{Z}_2^4$ | $\mathbb{Z}_2^4$ | $\mathbb{Z}_2^4$ | $\mathbb{Z}_2^4$ | $\mathbb{Z}_2$ | $\mathbb{Z}\oplus\mathbb{Z}_2^3$ | $\mathbb{Z}\oplus\mathbb{Z}_2^3$ | $\mathbb{Z}_2^4$ | $\mathbb{Z}_2^4$ | $\mathbb{Z}_2^4$ |
| 27 | $\mathbb{Z}$ | $\mathbb{Z}_2^3$ | $\mathbb{Z}_2^4$ | $\mathbb{Z}_2^4$ | $\mathbb{Z}_2^4$ | $\mathbb{Z}_2^4$ | $\mathbb{Z}_2$ | $\mathbb{Z}\oplus\mathbb{Z}_2^3$ | $\mathbb{Z}\oplus\mathbb{Z}_2^3$ | $\mathbb{Z}_2^4$ | $\mathbb{Z}_2^4$ | $\mathbb{Z}_2^4$ |
| 28 | $\mathbb{Z}$ | $\mathbb{Z}_2^3$ | $\mathbb{Z}_2^4$ | $\mathbb{Z}_2^4$ | $\mathbb{Z}_2^4$ | $\mathbb{Z}_2^4$ | $\mathbb{Z}_2$ | $\mathbb{Z}\oplus\mathbb{Z}_2^3$ | $\mathbb{Z}\oplus\mathbb{Z}_2^3$ | $\mathbb{Z}_2^4$ | $\mathbb{Z}_2^4$ | $\mathbb{Z}_2^4$ |
| 29 | $\mathbb{Z}$ | $\mathbb{Z}_2^2$ | $\mathbb{Z}_2$ | $\mathbb{Z}_1$ | $\mathbb{Z}_1$ | $\mathbb{Z}_1$ | $\mathbb{Z}_2$ | $\mathbb{Z}\oplus\mathbb{Z}_2^2$ | $\mathbb{Z}$ | $\mathbb{Z}_1$ | $\mathbb{Z}_1$ | $\mathbb{Z}_1$ |
| 30 | $\mathbb{Z}$ | $\mathbb{Z}_2^2$ | $\mathbb{Z}_2\oplus\mathbb{Z}_4$ | $\mathbb{Z}_2^2$ | $\mathbb{Z}_2^2$ | $\mathbb{Z}_2^2$ | $\mathbb{Z}_2$ | $\mathbb{Z}\oplus\mathbb{Z}_2^2$ | $\mathbb{Z}\oplus\mathbb{Z}_2$ | $\mathbb{Z}_2^2$ | $\mathbb{Z}_2^2$ | $\mathbb{Z}_2^2$ |
| 31 | $\mathbb{Z}$ | $\mathbb{Z}_2^2$ | $\mathbb{Z}_2^2$ | $\mathbb{Z}_2^2$ | $\mathbb{Z}_2^2$ | $\mathbb{Z}_2^2$ | $\mathbb{Z}_2$ | $\mathbb{Z}\oplus\mathbb{Z}_2\oplus\mathbb{Z}_4$ | $\mathbb{Z}\oplus\mathbb{Z}_2$ | $\mathbb{Z}_2^2$ | $\mathbb{Z}_2^2$ | $\mathbb{Z}_2^2$ |
| 32 | $\mathbb{Z}$ | $\mathbb{Z}_2\oplus\mathbb{Z}_4$ | $\mathbb{Z}_2\oplus\mathbb{Z}_4$ | $\mathbb{Z}_2^2$ | $\mathbb{Z}_2^2$ | $\mathbb{Z}_2^2$ | $\mathbb{Z}_2$ | $\mathbb{Z}\oplus\mathbb{Z}_2^2$ | $\mathbb{Z}\oplus\mathbb{Z}_2$ | $\mathbb{Z}_2^2$ | $\mathbb{Z}_2^2$ | $\mathbb{Z}_2^2$ |
| 33 | $\mathbb{Z}$ | $\mathbb{Z}_4$ | $\mathbb{Z}_2$ | $\mathbb{Z}_1$ | $\mathbb{Z}_1$ | $\mathbb{Z}_1$ | $\mathbb{Z}_2$ | $\mathbb{Z}\oplus\mathbb{Z}_4$ | $\mathbb{Z}$ | $\mathbb{Z}_1$ | $\mathbb{Z}_1$ | $\mathbb{Z}_1$ |
| 34 | $\mathbb{Z}$ | $\mathbb{Z}_2^2$ | $\mathbb{Z}_2\oplus\mathbb{Z}_4$ | $\mathbb{Z}_2^2$ | $\mathbb{Z}_2^2$ | $\mathbb{Z}_2^2$ | $\mathbb{Z}_2$ | $\mathbb{Z}\oplus\mathbb{Z}_2^2$ | $\mathbb{Z}\oplus\mathbb{Z}_2$ | $\mathbb{Z}_2^2$ | $\mathbb{Z}_2^2$ | $\mathbb{Z}_2^2$ |
| 35 | $\mathbb{Z}$ | $\mathbb{Z}_2^3$ | $\mathbb{Z}_2^5$ | $\mathbb{Z}_2^7$ | $\mathbb{Z}_2^9$ | $\mathbb{Z}_2^{11}$ | $\mathbb{Z}_2$ | $\mathbb{Z}\oplus\mathbb{Z}_2^3$ | $\mathbb{Z}\oplus\mathbb{Z}_2^4$ | $\mathbb{Z}_2^7$ | $\mathbb{Z}_2^9$ | $\mathbb{Z}_2^{11}$ |
| 36 | $\mathbb{Z}$ | $\mathbb{Z}_2^2$ | $\mathbb{Z}_2^2$ | $\mathbb{Z}_2^2$ | $\mathbb{Z}_2^2$ | $\mathbb{Z}_2^2$ | $\mathbb{Z}_2$ | $\mathbb{Z}\oplus\mathbb{Z}_2^2$ | $\mathbb{Z}\oplus\mathbb{Z}_2$ | $\mathbb{Z}_2^2$ | $\mathbb{Z}_2^2$ | $\mathbb{Z}_2^2$ |
| 37 | $\mathbb{Z}$ | $\mathbb{Z}_2^2$ | $\mathbb{Z}_2^3$ | $\mathbb{Z}_2^3$ | $\mathbb{Z}_2^3$ | $\mathbb{Z}_2^3$ | $\mathbb{Z}_2$ | $\mathbb{Z}\oplus\mathbb{Z}_2^2$ | $\mathbb{Z}\oplus\mathbb{Z}_2^2$ | $\mathbb{Z}_2^3$ | $\mathbb{Z}_2^3$ | $\mathbb{Z}_2^3$ |
| 38 | $\mathbb{Z}$ | $\mathbb{Z}_2^3$ | $\mathbb{Z}_2^5$ | $\mathbb{Z}_2^7$ | $\mathbb{Z}_2^9$ | $\mathbb{Z}_2^{11}$ | $\mathbb{Z}_2$ | $\mathbb{Z}\oplus\mathbb{Z}_2^3$ | $\mathbb{Z}\oplus\mathbb{Z}_2^4$ | $\mathbb{Z}_2^7$ | $\mathbb{Z}_2^9$ | $\mathbb{Z}_2^{11}$ |
| 39 | $\mathbb{Z}$ | $\mathbb{Z}_2^3$ | $\mathbb{Z}_2^4$ | $\mathbb{Z}_2^4$ | $\mathbb{Z}_2^4$ | $\mathbb{Z}_2^4$ | $\mathbb{Z}_2$ | $\mathbb{Z}\oplus\mathbb{Z}_2^3$ | $\mathbb{Z}\oplus\mathbb{Z}_2^3$ | $\mathbb{Z}_2^4$ | $\mathbb{Z}_2^4$ | $\mathbb{Z}_2^4$ |
| 40 | $\mathbb{Z}$ | $\mathbb{Z}_2^2$ | $\mathbb{Z}_2^3$ | $\mathbb{Z}_2^3$ | $\mathbb{Z}_2^3$ | $\mathbb{Z}_2^3$ | $\mathbb{Z}_2$ | $\mathbb{Z}\oplus\mathbb{Z}_2^2$ | $\mathbb{Z}\oplus\mathbb{Z}_2^2$ | $\mathbb{Z}_2^3$ | $\mathbb{Z}_2^3$ | $\mathbb{Z}_2^3$ |
| 41 | $\mathbb{Z}$ | $\mathbb{Z}_2^2$ | $\mathbb{Z}_4$ | $\mathbb{Z}_2$ | $\mathbb{Z}_2$ | $\mathbb{Z}_2$ | $\mathbb{Z}_2$ | $\mathbb{Z}\oplus\mathbb{Z}_2^2$ | $\mathbb{Z}$ | $\mathbb{Z}_2$ | $\mathbb{Z}_2$ | $\mathbb{Z}_2$ |
| 42 | $\mathbb{Z}$ | $\mathbb{Z}_2^3$ | $\mathbb{Z}_2^4$ | $\mathbb{Z}_2^5$ | $\mathbb{Z}_2^6$ | $\mathbb{Z}_2^7$ | $\mathbb{Z}_2$ | $\mathbb{Z}\oplus\mathbb{Z}_2^3$ | $\mathbb{Z}\oplus\mathbb{Z}_2^3$ | $\mathbb{Z}_2^5$ | $\mathbb{Z}_2^6$ | $\mathbb{Z}_2^7$ |
| 43 | $\mathbb{Z}$ | $\mathbb{Z}_2$ | $\mathbb{Z}_4$ | $\mathbb{Z}_2$ | $\mathbb{Z}_2$ | $\mathbb{Z}_2$ | $\mathbb{Z}_2$ | $\mathbb{Z}\oplus\mathbb{Z}_2$ | $\mathbb{Z}$ | $\mathbb{Z}_2$ | $\mathbb{Z}_2$ | $\mathbb{Z}_2$ |
| 44 | $\mathbb{Z}$ | $\mathbb{Z}_2^2$ | $\mathbb{Z}_2^4$ | $\mathbb{Z}_2^6$ | $\mathbb{Z}_2^8$ | $\mathbb{Z}_2^{10}$ | $\mathbb{Z}_2$ | $\mathbb{Z}\oplus\mathbb{Z}_2\oplus\mathbb{Z}_4$ | $\mathbb{Z}\oplus\mathbb{Z}_2^3$ | $\mathbb{Z}_2^6$ | $\mathbb{Z}_2^8$ | $\mathbb{Z}_2^{10}$ |
| 45 | $\mathbb{Z}$ | $\mathbb{Z}_2^2$ | $\mathbb{Z}_2^2$ | $\mathbb{Z}_2^2$ | $\mathbb{Z}_2^2$ | $\mathbb{Z}_2^2$ | $\mathbb{Z}_2$ | $\mathbb{Z}\oplus\mathbb{Z}_2^2$ | $\mathbb{Z}\oplus\mathbb{Z}_2$ | $\mathbb{Z}_2^2$ | $\mathbb{Z}_2^2$ | $\mathbb{Z}_2^2$ |
| 46 | $\mathbb{Z}$ | $\mathbb{Z}_2^2$ | $\mathbb{Z}_2^3$ | $\mathbb{Z}_2^3$ | $\mathbb{Z}_2^3$ | $\mathbb{Z}_2^3$ | $\mathbb{Z}_2$ | $\mathbb{Z}\oplus\mathbb{Z}_2^2$ | $\mathbb{Z}\oplus\mathbb{Z}_2^2$ | $\mathbb{Z}_2^3$ | $\mathbb{Z}_2^3$ | $\mathbb{Z}_2^3$ |
| 47 | $\mathbb{Z}_1$ | $\mathbb{Z}_2^6$ | $\mathbb{Z}_2^{12}$ | $\mathbb{Z}_2^{26}$ | $\mathbb{Z}_2^{40}$ | $\mathbb{Z}_2^{62}$ | $\mathbb{Z}_2$ | $\mathbb{Z}_2^5$ | $\mathbb{Z}\oplus\mathbb{Z}_2^{13}$ | $\mathbb{Z}_2^{24}$ | $\mathbb{Z}_2^{42}$ | $\mathbb{Z}_2^{60}$ |
| 48 | $\mathbb{Z}_1$ | $\mathbb{Z}_2^4$ | $\mathbb{Z}_2^3\oplus\mathbb{Z}_4$ | $\mathbb{Z}_2^8$ | $\mathbb{Z}_2^8$ | $\mathbb{Z}_2^{12}$ | $\mathbb{Z}_2$ | $\mathbb{Z}_2^3$ | $\mathbb{Z}\oplus\mathbb{Z}_2^5$ | $\mathbb{Z}_2^6$ | $\mathbb{Z}_2^{10}$ | $\mathbb{Z}_2^{10}$ |
| 49 | $\mathbb{Z}_1$ | $\mathbb{Z}_2^5$ | $\mathbb{Z}_2^7$ | $\mathbb{Z}_2^{13}$ | $\mathbb{Z}_2^{15}$ | $\mathbb{Z}_2^{21}$ | $\mathbb{Z}_2$ | $\mathbb{Z}_2^4$ | $\mathbb{Z}\oplus\mathbb{Z}_2^8$ | $\mathbb{Z}_2^{11}$ | $\mathbb{Z}_2^{17}$ | $\mathbb{Z}_2^{19}$ |
| 50 | $\mathbb{Z}_1$ | $\mathbb{Z}_2^4$ | $\mathbb{Z}_2^3\oplus\mathbb{Z}_4$ | $\mathbb{Z}_2^8$ | $\mathbb{Z}_2^8$ | $\mathbb{Z}_2^{12}$ | $\mathbb{Z}_2$ | $\mathbb{Z}_2^3$ | $\mathbb{Z}\oplus\mathbb{Z}_2^5$ | $\mathbb{Z}_2^6$ | $\mathbb{Z}_2^{10}$ | $\mathbb{Z}_2^{10}$ |
| 51 | $\mathbb{Z}_1$ | $\mathbb{Z}_2^5$ | $\mathbb{Z}_2^7$ | $\mathbb{Z}_2^{13}$ | $\mathbb{Z}_2^{15}$ | $\mathbb{Z}_2^{21}$ | $\mathbb{Z}_2$ | $\mathbb{Z}_2^4$ | $\mathbb{Z}\oplus\mathbb{Z}_2^8$ | $\mathbb{Z}_2^{11}$ | $\mathbb{Z}_2^{17}$ | $\mathbb{Z}_2^{19}$ |
| 52 | $\mathbb{Z}_1$ | $\mathbb{Z}_2^3$ | $\mathbb{Z}_2^2$ | $\mathbb{Z}_2^4$ | $\mathbb{Z}_2^2$ | $\mathbb{Z}_2^4$ | $\mathbb{Z}_2$ | $\mathbb{Z}_2^2$ | $\mathbb{Z}\oplus\mathbb{Z}_2^3$ | $\mathbb{Z}_2^2$ | $\mathbb{Z}_2^4$ | $\mathbb{Z}_2^2$ |
| 53 | $\mathbb{Z}_1$ | $\mathbb{Z}_2^4$ | $\mathbb{Z}_2^4$ | $\mathbb{Z}_2^8$ | $\mathbb{Z}_2^8$ | $\mathbb{Z}_2^{12}$ | $\mathbb{Z}_2$ | $\mathbb{Z}_2^3$ | $\mathbb{Z}\oplus\mathbb{Z}_2^5$ | $\mathbb{Z}_2^6$ | $\mathbb{Z}_2^{10}$ | $\mathbb{Z}_2^{10}$ |
| 54 | $\mathbb{Z}_1$ | $\mathbb{Z}_2^4$ | $\mathbb{Z}_2^3$ | $\mathbb{Z}_2^5$ | $\mathbb{Z}_2^3$ | $\mathbb{Z}_2^5$ | $\mathbb{Z}_2$ | $\mathbb{Z}_2^3$ | $\mathbb{Z}\oplus\mathbb{Z}_2^4$ | $\mathbb{Z}_2^3$ | $\mathbb{Z}_2^5$ | $\mathbb{Z}_2^3$ |
| 55 | $\mathbb{Z}_1$ | $\mathbb{Z}_2^3\oplus\mathbb{Z}_4$ | $\mathbb{Z}_2^4$ | $\mathbb{Z}_2^8$ | $\mathbb{Z}_2^8$ | $\mathbb{Z}_2^{12}$ | $\mathbb{Z}_2$ | $\mathbb{Z}_2^3$ | $\mathbb{Z}\oplus\mathbb{Z}_2^5$ | $\mathbb{Z}_2^6$ | $\mathbb{Z}_2^{10}$ | $\mathbb{Z}_2^{10}$ |
| 56 | $\mathbb{Z}_1$ | $\mathbb{Z}_2^2\oplus\mathbb{Z}_4$ | $\mathbb{Z}_2^2$ | $\mathbb{Z}_2^4$ | $\mathbb{Z}_2^2$ | $\mathbb{Z}_2^4$ | $\mathbb{Z}_2$ | $\mathbb{Z}_2\oplus\mathbb{Z}_4$ | $\mathbb{Z}\oplus\mathbb{Z}_2^3$ | $\mathbb{Z}_2^2$ | $\mathbb{Z}_2^4$ | $\mathbb{Z}_2^2$ |

| # | | | | | | | | | | | | |
|---|---|---|---|---|---|---|---|---|---|---|---|---|
| 57 | $\mathbb{Z}_1$ | $\mathbb{Z}_2^4$ | $\mathbb{Z}_2^3$ | $\mathbb{Z}_2^5$ | $\mathbb{Z}_2^3$ | $\mathbb{Z}_2^5$ | $\mathbb{Z}_2$ | $\mathbb{Z}_2^3$ | $\mathbb{Z}\oplus\mathbb{Z}_2^4$ | $\mathbb{Z}_2^3$ | $\mathbb{Z}_2^5$ | $\mathbb{Z}_2^3$ |
| 58 | $\mathbb{Z}_1$ | $\mathbb{Z}_2^2\oplus\mathbb{Z}_4$ | $\mathbb{Z}_2^3$ | $\mathbb{Z}_2^7$ | $\mathbb{Z}_2^7$ | $\mathbb{Z}_2^{11}$ | $\mathbb{Z}_2$ | $\mathbb{Z}_2\oplus\mathbb{Z}_4$ | $\mathbb{Z}\oplus\mathbb{Z}_2^4$ | $\mathbb{Z}_2^5$ | $\mathbb{Z}_2^9$ | $\mathbb{Z}_2^9$ |
| 59 | $\mathbb{Z}_1$ | $\mathbb{Z}_2^4$ | $\mathbb{Z}_2^4$ | $\mathbb{Z}_2^8$ | $\mathbb{Z}_2^8$ | $\mathbb{Z}_2^{12}$ | $\mathbb{Z}_2$ | $\mathbb{Z}_2^2\oplus\mathbb{Z}_4$ | $\mathbb{Z}\oplus\mathbb{Z}_2^5$ | $\mathbb{Z}_2^6$ | $\mathbb{Z}_2^{10}$ | $\mathbb{Z}_2^{10}$ |
| 60 | $\mathbb{Z}_1$ | $\mathbb{Z}_2^3$ | $\mathbb{Z}_2$ | $\mathbb{Z}_2^3$ | $\mathbb{Z}_2$ | $\mathbb{Z}_2^3$ | $\mathbb{Z}_2$ | $\mathbb{Z}_2^2$ | $\mathbb{Z}\oplus\mathbb{Z}_2^2$ | $\mathbb{Z}_2$ | $\mathbb{Z}_2^3$ | $\mathbb{Z}_2$ |
| 61 | $\mathbb{Z}_1$ | $\mathbb{Z}_2^3$ | $\mathbb{Z}_1$ | $\mathbb{Z}_2^2$ | $\mathbb{Z}_1$ | $\mathbb{Z}_2^2$ | $\mathbb{Z}_2$ | $\mathbb{Z}_2^2$ | $\mathbb{Z}\oplus\mathbb{Z}_2$ | $\mathbb{Z}_1$ | $\mathbb{Z}_2^2$ | $\mathbb{Z}_1$ |
| 62 | $\mathbb{Z}_1$ | $\mathbb{Z}_2^2\oplus\mathbb{Z}_4$ | $\mathbb{Z}_2^2$ | $\mathbb{Z}_2^4$ | $\mathbb{Z}_2^2$ | $\mathbb{Z}_2^4$ | $\mathbb{Z}_2$ | $\mathbb{Z}_2\oplus\mathbb{Z}_4$ | $\mathbb{Z}\oplus\mathbb{Z}_2^3$ | $\mathbb{Z}_2^2$ | $\mathbb{Z}_2^4$ | $\mathbb{Z}_2^2$ |
| 63 | $\mathbb{Z}_1$ | $\mathbb{Z}_2^4$ | $\mathbb{Z}_2^4$ | $\mathbb{Z}_2^8$ | $\mathbb{Z}_2^8$ | $\mathbb{Z}_2^{12}$ | $\mathbb{Z}_2$ | $\mathbb{Z}_2^3$ | $\mathbb{Z}\oplus\mathbb{Z}_2^5$ | $\mathbb{Z}_2^6$ | $\mathbb{Z}_2^{10}$ | $\mathbb{Z}_2^{10}$ |
| 64 | $\mathbb{Z}_1$ | $\mathbb{Z}_2^4$ | $\mathbb{Z}_2^3$ | $\mathbb{Z}_2^6$ | $\mathbb{Z}_2^5$ | $\mathbb{Z}_2^8$ | $\mathbb{Z}_2$ | $\mathbb{Z}_2^3$ | $\mathbb{Z}\oplus\mathbb{Z}_2^4$ | $\mathbb{Z}_2^4$ | $\mathbb{Z}_2^7$ | $\mathbb{Z}_2^6$ |
| 65 | $\mathbb{Z}_1$ | $\mathbb{Z}_2^5$ | $\mathbb{Z}_2^8$ | $\mathbb{Z}_2^{17}$ | $\mathbb{Z}_2^{24}$ | $\mathbb{Z}_2^{37}$ | $\mathbb{Z}_2$ | $\mathbb{Z}_2^4$ | $\mathbb{Z}\oplus\mathbb{Z}_2^9$ | $\mathbb{Z}_2^{15}$ | $\mathbb{Z}_2^{26}$ | $\mathbb{Z}_2^{35}$ |
| 66 | $\mathbb{Z}_1$ | $\mathbb{Z}_2^4$ | $\mathbb{Z}_2^5$ | $\mathbb{Z}_2^{10}$ | $\mathbb{Z}_2^{11}$ | $\mathbb{Z}_2^{16}$ | $\mathbb{Z}_2$ | $\mathbb{Z}_2^3$ | $\mathbb{Z}\oplus\mathbb{Z}_2^6$ | $\mathbb{Z}_2^8$ | $\mathbb{Z}_2^{13}$ | $\mathbb{Z}_2^{14}$ |
| 67 | $\mathbb{Z}_1$ | $\mathbb{Z}_2^5$ | $\mathbb{Z}_2^7$ | $\mathbb{Z}_2^{13}$ | $\mathbb{Z}_2^{15}$ | $\mathbb{Z}_2^{21}$ | $\mathbb{Z}_2$ | $\mathbb{Z}_2^4$ | $\mathbb{Z}\oplus\mathbb{Z}_2^8$ | $\mathbb{Z}_2^{11}$ | $\mathbb{Z}_2^{17}$ | $\mathbb{Z}_2^{19}$ |
| 68 | $\mathbb{Z}_1$ | $\mathbb{Z}_2^4$ | $\mathbb{Z}_2^2\oplus\mathbb{Z}_4$ | $\mathbb{Z}_2^6$ | $\mathbb{Z}_2^5$ | $\mathbb{Z}_2^8$ | $\mathbb{Z}_2$ | $\mathbb{Z}_2^3$ | $\mathbb{Z}\oplus\mathbb{Z}_2^4$ | $\mathbb{Z}_2^4$ | $\mathbb{Z}_2^7$ | $\mathbb{Z}_2^6$ |
| 69 | $\mathbb{Z}_1$ | $\mathbb{Z}_2^5$ | $\mathbb{Z}_2^7$ | $\mathbb{Z}_2^{14}$ | $\mathbb{Z}_2^{18}$ | $\mathbb{Z}_2^{27}$ | $\mathbb{Z}_2$ | $\mathbb{Z}_2^4$ | $\mathbb{Z}\oplus\mathbb{Z}_2^8$ | $\mathbb{Z}_2^{12}$ | $\mathbb{Z}_2^{20}$ | $\mathbb{Z}_2^{25}$ |
| 70 | $\mathbb{Z}_1$ | $\mathbb{Z}_2^3$ | $\mathbb{Z}_2\oplus\mathbb{Z}_4$ | $\mathbb{Z}_2^5$ | $\mathbb{Z}_2^4$ | $\mathbb{Z}_2^7$ | $\mathbb{Z}_2$ | $\mathbb{Z}_2^2$ | $\mathbb{Z}\oplus\mathbb{Z}_2^3$ | $\mathbb{Z}_2^3$ | $\mathbb{Z}_2^6$ | $\mathbb{Z}_2^5$ |
| 71 | $\mathbb{Z}_1$ | $\mathbb{Z}_2^4$ | $\mathbb{Z}_2^6$ | $\mathbb{Z}_2^{14}$ | $\mathbb{Z}_2^{20}$ | $\mathbb{Z}_2^{32}$ | $\mathbb{Z}_2$ | $\mathbb{Z}_2^2\oplus\mathbb{Z}_4$ | $\mathbb{Z}\oplus\mathbb{Z}_2^7$ | $\mathbb{Z}_2^{12}$ | $\mathbb{Z}_2^{22}$ | $\mathbb{Z}_2^{30}$ |
| 72 | $\mathbb{Z}_1$ | $\mathbb{Z}_2^4$ | $\mathbb{Z}_2^4$ | $\mathbb{Z}_2^8$ | $\mathbb{Z}_2^8$ | $\mathbb{Z}_2^{12}$ | $\mathbb{Z}_2$ | $\mathbb{Z}_2^3$ | $\mathbb{Z}\oplus\mathbb{Z}_2^5$ | $\mathbb{Z}_2^6$ | $\mathbb{Z}_2^{10}$ | $\mathbb{Z}_2^{10}$ |
| 73 | $\mathbb{Z}_1$ | $\mathbb{Z}_2^4$ | $\mathbb{Z}_2^3$ | $\mathbb{Z}_2^5$ | $\mathbb{Z}_2^3$ | $\mathbb{Z}_2^5$ | $\mathbb{Z}_2$ | $\mathbb{Z}_2^3$ | $\mathbb{Z}\oplus\mathbb{Z}_2^4$ | $\mathbb{Z}_2^3$ | $\mathbb{Z}_2^5$ | $\mathbb{Z}_2^3$ |
| 74 | $\mathbb{Z}_1$ | $\mathbb{Z}_2^4$ | $\mathbb{Z}_2^5$ | $\mathbb{Z}_2^{10}$ | $\mathbb{Z}_2^{11}$ | $\mathbb{Z}_2^{16}$ | $\mathbb{Z}_2$ | $\mathbb{Z}_2^3$ | $\mathbb{Z}\oplus\mathbb{Z}_2^6$ | $\mathbb{Z}_2^8$ | $\mathbb{Z}_2^{13}$ | $\mathbb{Z}_2^{14}$ |
| 75 | $\mathbb{Z}$ | $\mathbb{Z}\oplus\mathbb{Z}_2\oplus\mathbb{Z}_4$ | $\mathbb{Z}\oplus\mathbb{Z}_2\oplus\mathbb{Z}_4$ | $\mathbb{Z}_2\oplus\mathbb{Z}_4^2$ | $\mathbb{Z}_2\oplus\mathbb{Z}_4^2$ | $\mathbb{Z}_2\oplus\mathbb{Z}_4^2$ | $\mathbb{Z}$ | $\mathbb{Z}\oplus\mathbb{Z}_2\oplus\mathbb{Z}_4$ | $\mathbb{Z}\oplus\mathbb{Z}_2\oplus\mathbb{Z}_4$ | $\mathbb{Z}_2\oplus\mathbb{Z}_4^2$ | $\mathbb{Z}_2\oplus\mathbb{Z}_4^2$ | $\mathbb{Z}_2\oplus\mathbb{Z}_4^2$ |
| 76 | $\mathbb{Z}$ | $\mathbb{Z}\oplus\mathbb{Z}_2$ | $\mathbb{Z}$ | $\mathbb{Z}_1$ | $\mathbb{Z}_1$ | $\mathbb{Z}_1$ | $\mathbb{Z}$ | $\mathbb{Z}\oplus\mathbb{Z}_2$ | $\mathbb{Z}$ | $\mathbb{Z}_1$ | $\mathbb{Z}_1$ | $\mathbb{Z}_1$ |
| 77 | $\mathbb{Z}$ | $\mathbb{Z}\oplus\mathbb{Z}_2^2$ | $\mathbb{Z}\oplus\mathbb{Z}_2^2$ | $\mathbb{Z}_2^3$ | $\mathbb{Z}_2^3$ | $\mathbb{Z}_2^3$ | $\mathbb{Z}$ | $\mathbb{Z}\oplus\mathbb{Z}_2^2$ | $\mathbb{Z}\oplus\mathbb{Z}_2^2$ | $\mathbb{Z}_2^3$ | $\mathbb{Z}_2^3$ | $\mathbb{Z}_2^3$ |
| 78 | $\mathbb{Z}$ | $\mathbb{Z}\oplus\mathbb{Z}_2$ | $\mathbb{Z}$ | $\mathbb{Z}_1$ | $\mathbb{Z}_1$ | $\mathbb{Z}_1$ | $\mathbb{Z}$ | $\mathbb{Z}\oplus\mathbb{Z}_2$ | $\mathbb{Z}$ | $\mathbb{Z}_1$ | $\mathbb{Z}_1$ | $\mathbb{Z}_1$ |
| 79 | $\mathbb{Z}$ | $\mathbb{Z}\oplus\mathbb{Z}_4$ | $\mathbb{Z}\oplus\mathbb{Z}_2$ | $\mathbb{Z}_2\oplus\mathbb{Z}_4$ | $\mathbb{Z}_2\oplus\mathbb{Z}_4$ | $\mathbb{Z}_2\oplus\mathbb{Z}_4$ | $\mathbb{Z}$ | $\mathbb{Z}\oplus\mathbb{Z}_4$ | $\mathbb{Z}\oplus\mathbb{Z}_2$ | $\mathbb{Z}_2\oplus\mathbb{Z}_4$ | $\mathbb{Z}_2\oplus\mathbb{Z}_4$ | $\mathbb{Z}_2\oplus\mathbb{Z}_4$ |
| 80 | $\mathbb{Z}$ | $\mathbb{Z}\oplus\mathbb{Z}_2$ | $\mathbb{Z}$ | $\mathbb{Z}_2$ | $\mathbb{Z}_2$ | $\mathbb{Z}_2$ | $\mathbb{Z}$ | $\mathbb{Z}\oplus\mathbb{Z}_2$ | $\mathbb{Z}$ | $\mathbb{Z}_2$ | $\mathbb{Z}_2$ | $\mathbb{Z}_2$ |
| 81 | $\mathbb{Z}_1$ | $\mathbb{Z}\oplus\mathbb{Z}_2^3\oplus\mathbb{Z}_4$ | $\mathbb{Z}_2$ | $\mathbb{Z}_2^3\oplus\mathbb{Z}_4^2$ | $\mathbb{Z}_2$ | $\mathbb{Z}_2^3\oplus\mathbb{Z}_4^2$ | $\mathbb{Z}\oplus\mathbb{Z}_2$ | $\mathbb{Z}_2$ | $\mathbb{Z}\oplus\mathbb{Z}_2^3\oplus\mathbb{Z}_4$ | $\mathbb{Z}_2$ | $\mathbb{Z}_2^3\oplus\mathbb{Z}_4^2$ | $\mathbb{Z}_2$ |
| 82 | $\mathbb{Z}_1$ | $\mathbb{Z}\oplus\mathbb{Z}_4^2$ | $\mathbb{Z}_1$ | $\mathbb{Z}_2^2\oplus\mathbb{Z}_4^2$ | $\mathbb{Z}_1$ | $\mathbb{Z}_2^2\oplus\mathbb{Z}_4^2$ | $\mathbb{Z}\oplus\mathbb{Z}_2$ | $\mathbb{Z}_1$ | $\mathbb{Z}\oplus\mathbb{Z}_2^2\oplus\mathbb{Z}_4$ | $\mathbb{Z}_1$ | $\mathbb{Z}_2^2\oplus\mathbb{Z}_4^2$ | $\mathbb{Z}_1$ |
| 83 | $\mathbb{Z}_1$ | $\mathbb{Z}\oplus\mathbb{Z}_2^3\oplus\mathbb{Z}_4$ | $\mathbb{Z}_2^4$ | $\mathbb{Z}_2^9\oplus\mathbb{Z}_4^2$ | $\mathbb{Z}_2^{10}$ | $\mathbb{Z}_2^{15}\oplus\mathbb{Z}_4^2$ | $\mathbb{Z}\oplus\mathbb{Z}_2$ | $\mathbb{Z}_2^2$ | $\mathbb{Z}\oplus\mathbb{Z}_2^6\oplus\mathbb{Z}_4$ | $\mathbb{Z}_2^7$ | $\mathbb{Z}_2^{12}\oplus\mathbb{Z}_4^2$ | $\mathbb{Z}_2^{13}$ |
| 84 | $\mathbb{Z}_1$ | $\mathbb{Z}\oplus\mathbb{Z}_2^2\oplus\mathbb{Z}_4$ | $\mathbb{Z}_2^3$ | $\mathbb{Z}_2^6\oplus\mathbb{Z}_4^2$ | $\mathbb{Z}_2^7$ | $\mathbb{Z}_2^{10}\oplus\mathbb{Z}_4^2$ | $\mathbb{Z}\oplus\mathbb{Z}_2$ | $\mathbb{Z}_2^2$ | $\mathbb{Z}\oplus\mathbb{Z}_2^6$ | $\mathbb{Z}_2^4$ | $\mathbb{Z}_2^{11}$ | $\mathbb{Z}_2^8$ |
| 85 | $\mathbb{Z}_1$ | $\mathbb{Z}\oplus\mathbb{Z}_2^2\oplus\mathbb{Z}_4$ | $\mathbb{Z}_4$ | $\mathbb{Z}_2^3\oplus\mathbb{Z}_4^2$ | $\mathbb{Z}_4$ | $\mathbb{Z}_2^3\oplus\mathbb{Z}_4^2$ | $\mathbb{Z}\oplus\mathbb{Z}_2$ | $\mathbb{Z}_4$ | $\mathbb{Z}\oplus\mathbb{Z}_2^3\oplus\mathbb{Z}_4$ | $\mathbb{Z}_4$ | $\mathbb{Z}_2^3\oplus\mathbb{Z}_4^2$ | $\mathbb{Z}_4$ |
| 86 | $\mathbb{Z}_1$ | $\mathbb{Z}\oplus\mathbb{Z}_2^2\oplus\mathbb{Z}_4$ | $\mathbb{Z}_2$ | $\mathbb{Z}_2^4\oplus\mathbb{Z}_4$ | $\mathbb{Z}_2$ | $\mathbb{Z}_2^4\oplus\mathbb{Z}_4$ | $\mathbb{Z}\oplus\mathbb{Z}_2$ | $\mathbb{Z}_2$ | $\mathbb{Z}\oplus\mathbb{Z}_2^4$ | $\mathbb{Z}_2$ | $\mathbb{Z}_2^4\oplus\mathbb{Z}_4$ | $\mathbb{Z}_2$ |
| 87 | $\mathbb{Z}_1$ | $\mathbb{Z}\oplus\mathbb{Z}_2^2\oplus\mathbb{Z}_4$ | $\mathbb{Z}_2^2$ | $\mathbb{Z}_2^5\oplus\mathbb{Z}_4^2$ | $\mathbb{Z}_2^5$ | $\mathbb{Z}_2^8\oplus\mathbb{Z}_4^2$ | $\mathbb{Z}\oplus\mathbb{Z}_2$ | $\mathbb{Z}_2$ | $\mathbb{Z}\oplus\mathbb{Z}_2^5$ | $\mathbb{Z}_2^3$ | $\mathbb{Z}_2^8\oplus\mathbb{Z}_4$ | $\mathbb{Z}_2^6$ |
| 88 | $\mathbb{Z}_1$ | $\mathbb{Z}\oplus\mathbb{Z}_2\oplus\mathbb{Z}_4$ | $\mathbb{Z}_1$ | $\mathbb{Z}_2^3\oplus\mathbb{Z}_4$ | $\mathbb{Z}_1$ | $\mathbb{Z}_2^3\oplus\mathbb{Z}_4$ | $\mathbb{Z}\oplus\mathbb{Z}_2$ | $\mathbb{Z}_1$ | $\mathbb{Z}\oplus\mathbb{Z}_2^3$ | $\mathbb{Z}_1$ | $\mathbb{Z}_2^3\oplus\mathbb{Z}_4$ | $\mathbb{Z}_1$ |
| 89 | $\mathbb{Z}_1$ | $\mathbb{Z}_2^4$ | $\mathbb{Z}\oplus\mathbb{Z}_2^4\oplus\mathbb{Z}_4$ | $\mathbb{Z}_2^7\oplus\mathbb{Z}_4^2$ | $\mathbb{Z}_2^{12}$ | $\mathbb{Z}_2^{15}$ | $\mathbb{Z}_1$ | $\mathbb{Z}_2^4$ | $\mathbb{Z}\oplus\mathbb{Z}_2^4\oplus\mathbb{Z}_4$ | $\mathbb{Z}_2^7\oplus\mathbb{Z}_4^2$ | $\mathbb{Z}_2^{12}$ | $\mathbb{Z}_2^{15}$ |
| 90 | $\mathbb{Z}_1$ | $\mathbb{Z}_2^2\oplus\mathbb{Z}_4$ | $\mathbb{Z}\oplus\mathbb{Z}_2\oplus\mathbb{Z}_4$ | $\mathbb{Z}_2^3\oplus\mathbb{Z}_4$ | $\mathbb{Z}_2^4\oplus\mathbb{Z}_4$ | $\mathbb{Z}_2^5\oplus\mathbb{Z}_4$ | $\mathbb{Z}_1$ | $\mathbb{Z}_2^2\oplus\mathbb{Z}_4$ | $\mathbb{Z}\oplus\mathbb{Z}_2\oplus\mathbb{Z}_4$ | $\mathbb{Z}_2^3\oplus\mathbb{Z}_4$ | $\mathbb{Z}_2^4\oplus\mathbb{Z}_4$ | $\mathbb{Z}_2^5\oplus\mathbb{Z}_4$ |
| 91 | $\mathbb{Z}_1$ | $\mathbb{Z}_2^3$ | $\mathbb{Z}\oplus\mathbb{Z}_2^2$ | $\mathbb{Z}_2^3$ | $\mathbb{Z}_2^3$ | $\mathbb{Z}_2^3$ | $\mathbb{Z}_1$ | $\mathbb{Z}_2^3$ | $\mathbb{Z}\oplus\mathbb{Z}_2^2$ | $\mathbb{Z}_2^3$ | $\mathbb{Z}_2^3$ | $\mathbb{Z}_2^3$ |
| 92 | $\mathbb{Z}_1$ | $\mathbb{Z}_2\oplus\mathbb{Z}_4$ | $\mathbb{Z}$ | $\mathbb{Z}_2$ | $\mathbb{Z}_2$ | $\mathbb{Z}_2$ | $\mathbb{Z}_1$ | $\mathbb{Z}_2\oplus\mathbb{Z}_4$ | $\mathbb{Z}$ | $\mathbb{Z}_2$ | $\mathbb{Z}_2$ | $\mathbb{Z}_2$ |
| 93 | $\mathbb{Z}_1$ | $\mathbb{Z}_2^4$ | $\mathbb{Z}\oplus\mathbb{Z}_2^5$ | $\mathbb{Z}_2^9$ | $\mathbb{Z}_2^{12}$ | $\mathbb{Z}_2^{15}$ | $\mathbb{Z}_1$ | $\mathbb{Z}_2^4$ | $\mathbb{Z}\oplus\mathbb{Z}_2^5$ | $\mathbb{Z}_2^9$ | $\mathbb{Z}_2^{12}$ | $\mathbb{Z}_2^{15}$ |
| 94 | $\mathbb{Z}_1$ | $\mathbb{Z}_2^2\oplus\mathbb{Z}_4$ | $\mathbb{Z}\oplus\mathbb{Z}_2^2$ | $\mathbb{Z}_2^4$ | $\mathbb{Z}_2^5$ | $\mathbb{Z}_2^6$ | $\mathbb{Z}_1$ | $\mathbb{Z}_2^2\oplus\mathbb{Z}_4$ | $\mathbb{Z}\oplus\mathbb{Z}_2^2$ | $\mathbb{Z}_2^4$ | $\mathbb{Z}_2^5$ | $\mathbb{Z}_2^6$ |
| 95 | $\mathbb{Z}_1$ | $\mathbb{Z}_2^3$ | $\mathbb{Z}\oplus\mathbb{Z}_2^2$ | $\mathbb{Z}_2^3$ | $\mathbb{Z}_2^3$ | $\mathbb{Z}_2^3$ | $\mathbb{Z}_1$ | $\mathbb{Z}_2^3$ | $\mathbb{Z}\oplus\mathbb{Z}_2^2$ | $\mathbb{Z}_2^3$ | $\mathbb{Z}_2^3$ | $\mathbb{Z}_2^3$ |
| 96 | $\mathbb{Z}_1$ | $\mathbb{Z}_2\oplus\mathbb{Z}_4$ | $\mathbb{Z}$ | $\mathbb{Z}_2$ | $\mathbb{Z}_2$ | $\mathbb{Z}_2$ | $\mathbb{Z}_1$ | $\mathbb{Z}_2\oplus\mathbb{Z}_4$ | $\mathbb{Z}$ | $\mathbb{Z}_2$ | $\mathbb{Z}_2$ | $\mathbb{Z}_2$ |
| 97 | $\mathbb{Z}_1$ | $\mathbb{Z}_2^3$ | $\mathbb{Z}\oplus\mathbb{Z}_2^3$ | $\mathbb{Z}_2^5\oplus\mathbb{Z}_4$ | $\mathbb{Z}_2^8$ | $\mathbb{Z}_2^{10}$ | $\mathbb{Z}_1$ | $\mathbb{Z}_2^3$ | $\mathbb{Z}\oplus\mathbb{Z}_2^3$ | $\mathbb{Z}_2^5\oplus\mathbb{Z}_4$ | $\mathbb{Z}_2^8$ | $\mathbb{Z}_2^{10}$ |
| 98 | $\mathbb{Z}_1$ | $\mathbb{Z}_2^3$ | $\mathbb{Z}\oplus\mathbb{Z}_2^2$ | $\mathbb{Z}_2^4$ | $\mathbb{Z}_2^5$ | $\mathbb{Z}_2^6$ | $\mathbb{Z}_1$ | $\mathbb{Z}_2^3$ | $\mathbb{Z}\oplus\mathbb{Z}_2^2$ | $\mathbb{Z}_2^4$ | $\mathbb{Z}_2^5$ | $\mathbb{Z}_2^6$ |
| 99 | $\mathbb{Z}$ | $\mathbb{Z}_2^3$ | $\mathbb{Z}_2^6$ | $\mathbb{Z}_2^7\oplus\mathbb{Z}_4^2$ | $\mathbb{Z}_2^{10}\oplus\mathbb{Z}_4^2$ | $\mathbb{Z}_2^{15}$ | $\mathbb{Z}_2$ | $\mathbb{Z}\oplus\mathbb{Z}_2^2\oplus\mathbb{Z}_4$ | $\mathbb{Z}\oplus\mathbb{Z}_2^4\oplus\mathbb{Z}_4$ | $\mathbb{Z}_2^9$ | $\mathbb{Z}_2^{12}$ | $\mathbb{Z}_2^{13}\oplus\mathbb{Z}_4^2$ |
| 100 | $\mathbb{Z}$ | $\mathbb{Z}_2\oplus\mathbb{Z}_4$ | $\mathbb{Z}_2^2\oplus\mathbb{Z}_4$ | $\mathbb{Z}_2^3\oplus\mathbb{Z}_4$ | $\mathbb{Z}_2^4\oplus\mathbb{Z}_4$ | $\mathbb{Z}_2^5\oplus\mathbb{Z}_4$ | $\mathbb{Z}_2$ | $\mathbb{Z}\oplus\mathbb{Z}_2\oplus\mathbb{Z}_4$ | $\mathbb{Z}\oplus\mathbb{Z}_2\oplus\mathbb{Z}_4$ | $\mathbb{Z}_2^3\oplus\mathbb{Z}_4$ | $\mathbb{Z}_2^4\oplus\mathbb{Z}_4$ | $\mathbb{Z}_2^5\oplus\mathbb{Z}_4$ |
| 101 | $\mathbb{Z}$ | $\mathbb{Z}_2^2$ | $\mathbb{Z}_2^4$ | $\mathbb{Z}_2^2\oplus\mathbb{Z}_4^2$ | $\mathbb{Z}_2^6$ | $\mathbb{Z}_2^6$ | $\mathbb{Z}_2$ | $\mathbb{Z}\oplus\mathbb{Z}_2\oplus\mathbb{Z}_4$ | $\mathbb{Z}\oplus\mathbb{Z}_2^3$ | $\mathbb{Z}_2^4$ | $\mathbb{Z}_2^6$ | $\mathbb{Z}_2^4\oplus\mathbb{Z}_4^2$ |
| 102 | $\mathbb{Z}$ | $\mathbb{Z}_2^2$ | $\mathbb{Z}_2^3$ | $\mathbb{Z}_2^4$ | $\mathbb{Z}_2^5$ | $\mathbb{Z}_2^6$ | $\mathbb{Z}_2$ | $\mathbb{Z}\oplus\mathbb{Z}_2\oplus\mathbb{Z}_4$ | $\mathbb{Z}\oplus\mathbb{Z}_2^2$ | $\mathbb{Z}_2^4$ | $\mathbb{Z}_2^5$ | $\mathbb{Z}_2^6$ |
| 103 | $\mathbb{Z}$ | $\mathbb{Z}_2^2$ | $\mathbb{Z}_2^3$ | $\mathbb{Z}_2\oplus\mathbb{Z}_4^2$ | $\mathbb{Z}_2\oplus\mathbb{Z}_4^2$ | $\mathbb{Z}_2^3$ | $\mathbb{Z}_2$ | $\mathbb{Z}\oplus\mathbb{Z}_2\oplus\mathbb{Z}_4$ | $\mathbb{Z}\oplus\mathbb{Z}_2\oplus\mathbb{Z}_4$ | $\mathbb{Z}_2^3$ | $\mathbb{Z}_2^3$ | $\mathbb{Z}_2\oplus\mathbb{Z}_4^2$ |
| 104 | $\mathbb{Z}$ | $\mathbb{Z}_4$ | $\mathbb{Z}_2\oplus\mathbb{Z}_4$ | $\mathbb{Z}_2\oplus\mathbb{Z}_4$ | $\mathbb{Z}_2\oplus\mathbb{Z}_4$ | $\mathbb{Z}_2\oplus\mathbb{Z}_4$ | $\mathbb{Z}_2$ | $\mathbb{Z}\oplus\mathbb{Z}_4$ | $\mathbb{Z}\oplus\mathbb{Z}_2$ | $\mathbb{Z}_2\oplus\mathbb{Z}_4$ | $\mathbb{Z}_2\oplus\mathbb{Z}_4$ | $\mathbb{Z}_2\oplus\mathbb{Z}_4$ |
| 105 | $\mathbb{Z}$ | $\mathbb{Z}_2^2$ | $\mathbb{Z}_2^5$ | $\mathbb{Z}_2^4\oplus\mathbb{Z}_4^2$ | $\mathbb{Z}_2^9$ | $\mathbb{Z}_2^{10}$ | $\mathbb{Z}_2$ | $\mathbb{Z}\oplus\mathbb{Z}_2\oplus\mathbb{Z}_4$ | $\mathbb{Z}\oplus\mathbb{Z}_2^4$ | $\mathbb{Z}_2^6$ | $\mathbb{Z}_2^9$ | $\mathbb{Z}_2^8\oplus\mathbb{Z}_4^2$ |
| 106 | $\mathbb{Z}$ | $\mathbb{Z}_4$ | $\mathbb{Z}_2^2$ | $\mathbb{Z}_2^2$ | $\mathbb{Z}_2^2$ | $\mathbb{Z}_2^2$ | $\mathbb{Z}_2$ | $\mathbb{Z}\oplus\mathbb{Z}_4$ | $\mathbb{Z}\oplus\mathbb{Z}_2$ | $\mathbb{Z}_2^2$ | $\mathbb{Z}_2^2$ | $\mathbb{Z}_2^2$ |
| 107 | $\mathbb{Z}$ | $\mathbb{Z}_2^2$ | $\mathbb{Z}_2^4$ | $\mathbb{Z}_2^3\oplus\mathbb{Z}_4^2$ | $\mathbb{Z}_2^6\oplus\mathbb{Z}_4$ | $\mathbb{Z}_2^8$ | $\mathbb{Z}_2$ | $\mathbb{Z}\oplus\mathbb{Z}_2\oplus\mathbb{Z}_4$ | $\mathbb{Z}\oplus\mathbb{Z}_2^3$ | $\mathbb{Z}_2^5$ | $\mathbb{Z}_2^7$ | $\mathbb{Z}_2^6\oplus\mathbb{Z}_4^2$ |
| 108 | $\mathbb{Z}$ | $\mathbb{Z}_2^2$ | $\mathbb{Z}_2^3$ | $\mathbb{Z}_2\oplus\mathbb{Z}_4^2$ | $\mathbb{Z}_2^3\oplus\mathbb{Z}_4$ | $\mathbb{Z}_2^4$ | $\mathbb{Z}_2$ | $\mathbb{Z}\oplus\mathbb{Z}_2\oplus\mathbb{Z}_4$ | $\mathbb{Z}\oplus\mathbb{Z}_2\oplus\mathbb{Z}_4$ | $\mathbb{Z}_2^3$ | $\mathbb{Z}_2^4$ | $\mathbb{Z}_2^2\oplus\mathbb{Z}_4^2$ |
| 109 | $\mathbb{Z}$ | $\mathbb{Z}_2$ | $\mathbb{Z}_2^2$ | $\mathbb{Z}_2^3$ | $\mathbb{Z}_2^4$ | $\mathbb{Z}_2^5$ | $\mathbb{Z}_2$ | $\mathbb{Z}\oplus\mathbb{Z}_4$ | $\mathbb{Z}\oplus\mathbb{Z}_2$ | $\mathbb{Z}_2^3$ | $\mathbb{Z}_2^4$ | $\mathbb{Z}_2^5$ |
| 110 | $\mathbb{Z}$ | $\mathbb{Z}_2$ | $\mathbb{Z}_2$ | $\mathbb{Z}_2$ | $\mathbb{Z}_2$ | $\mathbb{Z}_2$ | $\mathbb{Z}_2$ | $\mathbb{Z}\oplus\mathbb{Z}_4$ | $\mathbb{Z}$ | $\mathbb{Z}_2$ | $\mathbb{Z}_2$ | $\mathbb{Z}_2$ |
| 111 | $\mathbb{Z}_1$ | $\mathbb{Z}_2^4$ | $\mathbb{Z}_2^5$ | $\mathbb{Z}_2^8\oplus\mathbb{Z}_4^2$ | $\mathbb{Z}_2^{11}$ | $\mathbb{Z}_2^{16}$ | $\mathbb{Z}_2$ | $\mathbb{Z}_2^3$ | $\mathbb{Z}\oplus\mathbb{Z}_2^5\oplus\mathbb{Z}_4$ | $\mathbb{Z}_2^8$ | $\mathbb{Z}_2^{13}$ | $\mathbb{Z}_2^{14}$ |
| 112 | $\mathbb{Z}_1$ | $\mathbb{Z}_2^2\oplus\mathbb{Z}_4$ | $\mathbb{Z}_2^4$ | $\mathbb{Z}_2^5\oplus\mathbb{Z}_4^2$ | $\mathbb{Z}_2^8$ | $\mathbb{Z}_2^9\oplus\mathbb{Z}_4^2$ | $\mathbb{Z}_2$ | $\mathbb{Z}_2^2$ | $\mathbb{Z}\oplus\mathbb{Z}_2^5$ | $\mathbb{Z}_2^5$ | $\mathbb{Z}_2^{10}$ | $\mathbb{Z}_2^9$ |
| 113 | $\mathbb{Z}_1$ | $\mathbb{Z}_2^2\oplus\mathbb{Z}_4$ | $\mathbb{Z}_2^2$ | $\mathbb{Z}_2^4\oplus\mathbb{Z}_4$ | $\mathbb{Z}_2^4$ | $\mathbb{Z}_2^6\oplus\mathbb{Z}_4$ | $\mathbb{Z}_2$ | $\mathbb{Z}_2\oplus\mathbb{Z}_4$ | $\mathbb{Z}\oplus\mathbb{Z}_2^2\oplus\mathbb{Z}_4$ | $\mathbb{Z}_2^3$ | $\mathbb{Z}_2^5\oplus\mathbb{Z}_4$ | $\mathbb{Z}_2^5$ |
| 114 | $\mathbb{Z}_1$ | $\mathbb{Z}_4^2$ | $\mathbb{Z}_2$ | $\mathbb{Z}_2^2\oplus\mathbb{Z}_4$ | $\mathbb{Z}_2$ | $\mathbb{Z}_2^2\oplus\mathbb{Z}_4$ | $\mathbb{Z}_2$ | $\mathbb{Z}_4$ | $\mathbb{Z}\oplus\mathbb{Z}_2^2$ | $\mathbb{Z}_2$ | $\mathbb{Z}_2^2\oplus\mathbb{Z}_4$ | $\mathbb{Z}_2$ |
| 115 | $\mathbb{Z}_1$ | $\mathbb{Z}_2^4$ | $\mathbb{Z}_2^5$ | $\mathbb{Z}_2^8\oplus\mathbb{Z}_4^2$ | $\mathbb{Z}_2^{11}$ | $\mathbb{Z}_2^{16}$ | $\mathbb{Z}_2$ | $\mathbb{Z}_2^3$ | $\mathbb{Z}\oplus\mathbb{Z}_2^5\oplus\mathbb{Z}_4$ | $\mathbb{Z}_2^8$ | $\mathbb{Z}_2^{13}$ | $\mathbb{Z}_2^{14}$ |
| 116 | $\mathbb{Z}_1$ | $\mathbb{Z}_2^2\oplus\mathbb{Z}_4$ | $\mathbb{Z}_2^3$ | $\mathbb{Z}_2^3\oplus\mathbb{Z}_4^2$ | $\mathbb{Z}_2^5$ | $\mathbb{Z}_2^5\oplus\mathbb{Z}_4^2$ | $\mathbb{Z}_2$ | $\mathbb{Z}_2^2$ | $\mathbb{Z}\oplus\mathbb{Z}_2^4$ | $\mathbb{Z}_2^3$ | $\mathbb{Z}_2^7$ | $\mathbb{Z}_2^5$ |
| 117 | $\mathbb{Z}_1$ | $\mathbb{Z}_2^2\oplus\mathbb{Z}_4$ | $\mathbb{Z}_2\oplus\mathbb{Z}_4$ | $\mathbb{Z}_2^4\oplus\mathbb{Z}_4$ | $\mathbb{Z}_2^4$ | $\mathbb{Z}_2^6\oplus\mathbb{Z}_4$ | $\mathbb{Z}_2$ | $\mathbb{Z}_2^2$ | $\mathbb{Z}\oplus\mathbb{Z}_2^2\oplus\mathbb{Z}_4$ | $\mathbb{Z}_2^3$ | $\mathbb{Z}_2^5\oplus\mathbb{Z}_4$ | $\mathbb{Z}_2^5$ |
| 118 | $\mathbb{Z}_1$ | $\mathbb{Z}_2^2\oplus\mathbb{Z}_4$ | $\mathbb{Z}_2\oplus\mathbb{Z}_4$ | $\mathbb{Z}_2^4\oplus\mathbb{Z}_4$ | $\mathbb{Z}_2^4$ | $\mathbb{Z}_2^6\oplus\mathbb{Z}_4$ | $\mathbb{Z}_2$ | $\mathbb{Z}_2^2$ | $\mathbb{Z}\oplus\mathbb{Z}_2^2\oplus\mathbb{Z}_4$ | $\mathbb{Z}_2^3$ | $\mathbb{Z}_2^5\oplus\mathbb{Z}_4$ | $\mathbb{Z}_2^5$ |
| 119 | $\mathbb{Z}_1$ | $\mathbb{Z}_2^3$ | $\mathbb{Z}_2^3$ | $\mathbb{Z}_2^5\oplus\mathbb{Z}_4^2$ | $\mathbb{Z}_2^7$ | $\mathbb{Z}_2^{11}$ | $\mathbb{Z}_2$ | $\mathbb{Z}_2^2$ | $\mathbb{Z}\oplus\mathbb{Z}_2^3\oplus\mathbb{Z}_4$ | $\mathbb{Z}_2^5$ | $\mathbb{Z}_2^9$ | $\mathbb{Z}_2^9$ |
| 120 | $\mathbb{Z}_1$ | $\mathbb{Z}_2^3$ | $\mathbb{Z}_2^2$ | $\mathbb{Z}_2^2\oplus\mathbb{Z}_4^2$ | $\mathbb{Z}_2^4$ | $\mathbb{Z}_2^4\oplus\mathbb{Z}_4^2$ | $\mathbb{Z}_2$ | $\mathbb{Z}_2^2$ | $\mathbb{Z}\oplus\mathbb{Z}_2^2\oplus\mathbb{Z}_4$ | $\mathbb{Z}_2^2$ | $\mathbb{Z}_2^6$ | $\mathbb{Z}_2^4$ |
| 121 | $\mathbb{Z}_1$ | $\mathbb{Z}_2^2\oplus\mathbb{Z}_4$ | $\mathbb{Z}_2^3$ | $\mathbb{Z}_2^4\oplus\mathbb{Z}_4^2$ | $\mathbb{Z}_2^6$ | $\mathbb{Z}_2^8\oplus\mathbb{Z}_4$ | $\mathbb{Z}_2$ | $\mathbb{Z}_2^2$ | $\mathbb{Z}\oplus\mathbb{Z}_2^4$ | $\mathbb{Z}_2^4$ | $\mathbb{Z}_2^8$ | $\mathbb{Z}_2^7$ |

| | | | | | | | | | | | | |
|---|---|---|---|---|---|---|---|---|---|---|---|---|
| 122 | $\mathbb{Z}_1$ | $\mathbb{Z}_2\oplus\mathbb{Z}_4$ | $\mathbb{Z}_2$ | $\mathbb{Z}_2^2\oplus\mathbb{Z}_4$ | $\mathbb{Z}_2$ | $\mathbb{Z}_2^2\oplus\mathbb{Z}_4$ | $\mathbb{Z}_2$ | $\mathbb{Z}_2$ | $\mathbb{Z}\oplus\mathbb{Z}_2^2$ | $\mathbb{Z}_2$ | $\mathbb{Z}_2^2\oplus\mathbb{Z}_4$ | $\mathbb{Z}_2$ |
| 123 | $\mathbb{Z}_1$ | $\mathbb{Z}_2^5$ | $\mathbb{Z}_2^9$ | $\mathbb{Z}_2^{18}\oplus\mathbb{Z}_4^2$ | $\mathbb{Z}_2^{30}$ | $\mathbb{Z}_2^{47}$ | $\mathbb{Z}_2$ | $\mathbb{Z}_2^4$ | $\mathbb{Z}\oplus\mathbb{Z}_2^9\oplus\mathbb{Z}_4$ | $\mathbb{Z}_2^{18}$ | $\mathbb{Z}_2^{32}$ | $\mathbb{Z}_2^{45}$ |
| 124 | $\mathbb{Z}_1$ | $\mathbb{Z}_2^4$ | $\mathbb{Z}_2^5$ | $\mathbb{Z}_2^8\oplus\mathbb{Z}_4^2$ | $\mathbb{Z}_2^{11}$ | $\mathbb{Z}_2^{16}$ | $\mathbb{Z}_2$ | $\mathbb{Z}_2^3$ | $\mathbb{Z}\oplus\mathbb{Z}_2^5\oplus\mathbb{Z}_4$ | $\mathbb{Z}_2^8$ | $\mathbb{Z}_2^{13}$ | $\mathbb{Z}_2^{14}$ |
| 125 | $\mathbb{Z}_1$ | $\mathbb{Z}_2^4$ | $\mathbb{Z}_2^4\oplus\mathbb{Z}_4$ | $\mathbb{Z}_2^8\oplus\mathbb{Z}_4^2$ | $\mathbb{Z}_2^{11}$ | $\mathbb{Z}_2^{16}$ | $\mathbb{Z}_2$ | $\mathbb{Z}_2^3$ | $\mathbb{Z}\oplus\mathbb{Z}_2^5\oplus\mathbb{Z}_4$ | $\mathbb{Z}_2^7\oplus\mathbb{Z}_4$ | $\mathbb{Z}_2^{13}$ | $\mathbb{Z}_2^{14}$ |
| 126 | $\mathbb{Z}_1$ | $\mathbb{Z}_2^3$ | $\mathbb{Z}_2^2\oplus\mathbb{Z}_4$ | $\mathbb{Z}_2^4\oplus\mathbb{Z}_4^2$ | $\mathbb{Z}_2^6$ | $\mathbb{Z}_2^8\oplus\mathbb{Z}_4$ | $\mathbb{Z}_2$ | $\mathbb{Z}_2^2$ | $\mathbb{Z}\oplus\mathbb{Z}_2^4$ | $\mathbb{Z}_2^3\oplus\mathbb{Z}_4$ | $\mathbb{Z}_2^8$ | $\mathbb{Z}_2^7$ |
| 127 | $\mathbb{Z}_1$ | $\mathbb{Z}_2^3\oplus\mathbb{Z}_4$ | $\mathbb{Z}_2^5$ | $\mathbb{Z}_2^{10}\oplus\mathbb{Z}_4$ | $\mathbb{Z}_2^{14}$ | $\mathbb{Z}_2^{21}\oplus\mathbb{Z}_4$ | $\mathbb{Z}_2$ | $\mathbb{Z}_2^3$ | $\mathbb{Z}\oplus\mathbb{Z}_2^5\oplus\mathbb{Z}_4$ | $\mathbb{Z}_2^9$ | $\mathbb{Z}_2^{15}\oplus\mathbb{Z}_4$ | $\mathbb{Z}_2^{20}$ |
| 128 | $\mathbb{Z}_1$ | $\mathbb{Z}_2^2\oplus\mathbb{Z}_4$ | $\mathbb{Z}_2^3$ | $\mathbb{Z}_2^6\oplus\mathbb{Z}_4$ | $\mathbb{Z}_2^7$ | $\mathbb{Z}_2^{10}\oplus\mathbb{Z}_4$ | $\mathbb{Z}_2$ | $\mathbb{Z}_2^2$ | $\mathbb{Z}\oplus\mathbb{Z}_2^4$ | $\mathbb{Z}_2^5$ | $\mathbb{Z}_2^8\oplus\mathbb{Z}_4$ | $\mathbb{Z}_2^9$ |
| 129 | $\mathbb{Z}_1$ | $\mathbb{Z}_2^4$ | $\mathbb{Z}_2^5$ | $\mathbb{Z}_2^8\oplus\mathbb{Z}_4^2$ | $\mathbb{Z}_2^{10}\oplus\mathbb{Z}_4$ | $\mathbb{Z}_2^{16}$ | $\mathbb{Z}_2$ | $\mathbb{Z}_2^2\oplus\mathbb{Z}_4$ | $\mathbb{Z}\oplus\mathbb{Z}_2^5\oplus\mathbb{Z}_4$ | $\mathbb{Z}_2^8$ | $\mathbb{Z}_2^{13}$ | $\mathbb{Z}_2^{13}\oplus\mathbb{Z}_4$ |
| 130 | $\mathbb{Z}_1$ | $\mathbb{Z}_2^3$ | $\mathbb{Z}_2^2$ | $\mathbb{Z}_2^2\oplus\mathbb{Z}_4^2$ | $\mathbb{Z}_2^2\oplus\mathbb{Z}_4$ | $\mathbb{Z}_2^4\oplus\mathbb{Z}_4$ | $\mathbb{Z}_2$ | $\mathbb{Z}_2\oplus\mathbb{Z}_4$ | $\mathbb{Z}\oplus\mathbb{Z}_2^2\oplus\mathbb{Z}_4$ | $\mathbb{Z}_2^2$ | $\mathbb{Z}_2^5$ | $\mathbb{Z}_2^2\oplus\mathbb{Z}_4$ |
| 131 | $\mathbb{Z}_1$ | $\mathbb{Z}_2^4$ | $\mathbb{Z}_2^7$ | $\mathbb{Z}_2^{13}\oplus\mathbb{Z}_4^2$ | $\mathbb{Z}_2^{22}$ | $\mathbb{Z}_2^{34}$ | $\mathbb{Z}_2$ | $\mathbb{Z}_2^3$ | $\mathbb{Z}\oplus\mathbb{Z}_2^8$ | $\mathbb{Z}_2^{13}$ | $\mathbb{Z}_2^{24}$ | $\mathbb{Z}_2^{32}$ |
| 132 | $\mathbb{Z}_1$ | $\mathbb{Z}_2^4$ | $\mathbb{Z}_2^6$ | $\mathbb{Z}_2^{10}\oplus\mathbb{Z}_4^2$ | $\mathbb{Z}_2^{16}$ | $\mathbb{Z}_2^{24}$ | $\mathbb{Z}_2$ | $\mathbb{Z}_2^3$ | $\mathbb{Z}\oplus\mathbb{Z}_2^7$ | $\mathbb{Z}_2^{10}$ | $\mathbb{Z}_2^{18}$ | $\mathbb{Z}_2^{22}$ |
| 133 | $\mathbb{Z}_1$ | $\mathbb{Z}_2^3$ | $\mathbb{Z}_2^3$ | $\mathbb{Z}_2^5\oplus\mathbb{Z}_4$ | $\mathbb{Z}_2^6$ | $\mathbb{Z}_2^8\oplus\mathbb{Z}_4$ | $\mathbb{Z}_2$ | $\mathbb{Z}_2^2$ | $\mathbb{Z}\oplus\mathbb{Z}_2^4$ | $\mathbb{Z}_2^4$ | $\mathbb{Z}_2^8$ | $\mathbb{Z}_2^7$ |
| 134 | $\mathbb{Z}_1$ | $\mathbb{Z}_2^4$ | $\mathbb{Z}_2^5$ | $\mathbb{Z}_2^9\oplus\mathbb{Z}_4$ | $\mathbb{Z}_2^{11}$ | $\mathbb{Z}_2^{16}$ | $\mathbb{Z}_2$ | $\mathbb{Z}_2^3$ | $\mathbb{Z}\oplus\mathbb{Z}_2^6$ | $\mathbb{Z}_2^8$ | $\mathbb{Z}_2^{13}$ | $\mathbb{Z}_2^{14}$ |
| 135 | $\mathbb{Z}_1$ | $\mathbb{Z}_2^2\oplus\mathbb{Z}_4$ | $\mathbb{Z}_2^3$ | $\mathbb{Z}_2^5\oplus\mathbb{Z}_4$ | $\mathbb{Z}_2^6$ | $\mathbb{Z}_2^8\oplus\mathbb{Z}_4$ | $\mathbb{Z}_2$ | $\mathbb{Z}_2^2$ | $\mathbb{Z}\oplus\mathbb{Z}_2^4$ | $\mathbb{Z}_2^4$ | $\mathbb{Z}_2^8$ | $\mathbb{Z}_2^7$ |
| 136 | $\mathbb{Z}_1$ | $\mathbb{Z}_2^2\oplus\mathbb{Z}_4$ | $\mathbb{Z}_2^4$ | $\mathbb{Z}_2^8\oplus\mathbb{Z}_4$ | $\mathbb{Z}_2^{12}$ | $\mathbb{Z}_2^{18}\oplus\mathbb{Z}_4$ | $\mathbb{Z}_2$ | $\mathbb{Z}_2\oplus\mathbb{Z}_4$ | $\mathbb{Z}\oplus\mathbb{Z}_2^5$ | $\mathbb{Z}_2^7$ | $\mathbb{Z}_2^{14}$ | $\mathbb{Z}_2^{17}$ |
| 137 | $\mathbb{Z}_1$ | $\mathbb{Z}_2^3$ | $\mathbb{Z}_2^3$ | $\mathbb{Z}_2^4\oplus\mathbb{Z}_4^2$ | $\mathbb{Z}_2^6$ | $\mathbb{Z}_2^9$ | $\mathbb{Z}_2$ | $\mathbb{Z}_2\oplus\mathbb{Z}_4$ | $\mathbb{Z}\oplus\mathbb{Z}_2^4$ | $\mathbb{Z}_2^4$ | $\mathbb{Z}_2^8$ | $\mathbb{Z}_2^6\oplus\mathbb{Z}_4$ |
| 138 | $\mathbb{Z}_1$ | $\mathbb{Z}_2^2\oplus\mathbb{Z}_4$ | $\mathbb{Z}_2^4$ | $\mathbb{Z}_2^5\oplus\mathbb{Z}_4^2$ | $\mathbb{Z}_2^8$ | $\mathbb{Z}_2^{10}\oplus\mathbb{Z}_4$ | $\mathbb{Z}_2$ | $\mathbb{Z}_2\oplus\mathbb{Z}_4$ | $\mathbb{Z}\oplus\mathbb{Z}_2^5$ | $\mathbb{Z}_2^5$ | $\mathbb{Z}_2^{10}$ | $\mathbb{Z}_2^8\oplus\mathbb{Z}_4$ |
| 139 | $\mathbb{Z}_1$ | $\mathbb{Z}_2^4$ | $\mathbb{Z}_2^6$ | $\mathbb{Z}_2^{11}\oplus\mathbb{Z}_4^2$ | $\mathbb{Z}_2^{18}$ | $\mathbb{Z}_2^{28}$ | $\mathbb{Z}_2$ | $\mathbb{Z}_2^3$ | $\mathbb{Z}\oplus\mathbb{Z}_2^7$ | $\mathbb{Z}_2^{11}$ | $\mathbb{Z}_2^{20}$ | $\mathbb{Z}_2^{26}$ |
| 140 | $\mathbb{Z}_1$ | $\mathbb{Z}_2^4$ | $\mathbb{Z}_2^5$ | $\mathbb{Z}_2^8\oplus\mathbb{Z}_4^2$ | $\mathbb{Z}_2^{12}$ | $\mathbb{Z}_2^{18}$ | $\mathbb{Z}_2$ | $\mathbb{Z}_2^3$ | $\mathbb{Z}\oplus\mathbb{Z}_2^5\oplus\mathbb{Z}_4$ | $\mathbb{Z}_2^8$ | $\mathbb{Z}_2^{14}$ | $\mathbb{Z}_2^{16}$ |
| 141 | $\mathbb{Z}_1$ | $\mathbb{Z}_2^3$ | $\mathbb{Z}_2^3$ | $\mathbb{Z}_2^6\oplus\mathbb{Z}_4$ | $\mathbb{Z}_2^7$ | $\mathbb{Z}_2^{11}$ | $\mathbb{Z}_2$ | $\mathbb{Z}_2^2$ | $\mathbb{Z}\oplus\mathbb{Z}_2^4$ | $\mathbb{Z}_2^5$ | $\mathbb{Z}_2^9$ | $\mathbb{Z}_2^9$ |
| 142 | $\mathbb{Z}_1$ | $\mathbb{Z}_2^3$ | $\mathbb{Z}_2^2$ | $\mathbb{Z}_2^3\oplus\mathbb{Z}_4$ | $\mathbb{Z}_2^3$ | $\mathbb{Z}_2^4\oplus\mathbb{Z}_4$ | $\mathbb{Z}_2$ | $\mathbb{Z}_2^2$ | $\mathbb{Z}\oplus\mathbb{Z}_2^3$ | $\mathbb{Z}_2^2$ | $\mathbb{Z}_2^5$ | $\mathbb{Z}_2^3$ |
| 143 | $\mathbb{Z}$ | $\mathbb{Z}\oplus\mathbb{Z}_3^2$ | $\mathbb{Z}\oplus\mathbb{Z}_3^2$ | $\mathbb{Z}_3^3$ | $\mathbb{Z}_3^3$ | $\mathbb{Z}_3^3$ | $\mathbb{Z}$ | $\mathbb{Z}\oplus\mathbb{Z}_3^2$ | $\mathbb{Z}\oplus\mathbb{Z}_3^2$ | $\mathbb{Z}_3^3$ | $\mathbb{Z}_3^3$ | $\mathbb{Z}_3^3$ |
| 144 | $\mathbb{Z}$ | $\mathbb{Z}\oplus\mathbb{Z}_3$ | $\mathbb{Z}$ | $\mathbb{Z}_1$ | $\mathbb{Z}_1$ | $\mathbb{Z}_1$ | $\mathbb{Z}$ | $\mathbb{Z}\oplus\mathbb{Z}_3$ | $\mathbb{Z}$ | $\mathbb{Z}_1$ | $\mathbb{Z}_1$ | $\mathbb{Z}_1$ |
| 145 | $\mathbb{Z}$ | $\mathbb{Z}\oplus\mathbb{Z}_3$ | $\mathbb{Z}$ | $\mathbb{Z}_1$ | $\mathbb{Z}_1$ | $\mathbb{Z}_1$ | $\mathbb{Z}$ | $\mathbb{Z}\oplus\mathbb{Z}_3$ | $\mathbb{Z}$ | $\mathbb{Z}_1$ | $\mathbb{Z}_1$ | $\mathbb{Z}_1$ |
| 146 | $\mathbb{Z}$ | $\mathbb{Z}\oplus\mathbb{Z}_3$ | $\mathbb{Z}$ | $\mathbb{Z}_3$ | $\mathbb{Z}_3$ | $\mathbb{Z}_3$ | $\mathbb{Z}$ | $\mathbb{Z}\oplus\mathbb{Z}_3$ | $\mathbb{Z}$ | $\mathbb{Z}_3$ | $\mathbb{Z}_3$ | $\mathbb{Z}_3$ |
| 147 | $\mathbb{Z}_1$ | $\mathbb{Z}\oplus\mathbb{Z}_2^2\oplus\mathbb{Z}_3$ | $\mathbb{Z}_3$ | $\mathbb{Z}_2^4\oplus\mathbb{Z}_3^2$ | $\mathbb{Z}_3$ | $\mathbb{Z}_2^4\oplus\mathbb{Z}_3^2$ | $\mathbb{Z}\oplus\mathbb{Z}_2$ | $\mathbb{Z}_3$ | $\mathbb{Z}\oplus\mathbb{Z}_2^3\oplus\mathbb{Z}_3$ | $\mathbb{Z}_3$ | $\mathbb{Z}_2^4\oplus\mathbb{Z}_3^2$ | $\mathbb{Z}_3$ |
| 148 | $\mathbb{Z}_1$ | $\mathbb{Z}\oplus\mathbb{Z}_2^2\oplus\mathbb{Z}_3$ | $\mathbb{Z}_1$ | $\mathbb{Z}_2^4\oplus\mathbb{Z}_3$ | $\mathbb{Z}_1$ | $\mathbb{Z}_2^4\oplus\mathbb{Z}_3$ | $\mathbb{Z}\oplus\mathbb{Z}_2$ | $\mathbb{Z}_1$ | $\mathbb{Z}\oplus\mathbb{Z}_2^3$ | $\mathbb{Z}_1$ | $\mathbb{Z}_2^4\oplus\mathbb{Z}_3$ | $\mathbb{Z}_1$ |
| 149 | $\mathbb{Z}_1$ | $\mathbb{Z}_2^2$ | $\mathbb{Z}\oplus\mathbb{Z}_2\oplus\mathbb{Z}_3^2$ | $\mathbb{Z}_2^2\oplus\mathbb{Z}_3^3$ | $\mathbb{Z}_2^2$ | $\mathbb{Z}_2^2$ | $\mathbb{Z}_1$ | $\mathbb{Z}_2^2$ | $\mathbb{Z}\oplus\mathbb{Z}_2\oplus\mathbb{Z}_3^2$ | $\mathbb{Z}_2^2\oplus\mathbb{Z}_3^3$ | $\mathbb{Z}_2^2$ | $\mathbb{Z}_2^2$ |
| 150 | $\mathbb{Z}_1$ | $\mathbb{Z}_2^2\oplus\mathbb{Z}_3$ | $\mathbb{Z}\oplus\mathbb{Z}_2\oplus\mathbb{Z}_3$ | $\mathbb{Z}_2^2\oplus\mathbb{Z}_3^2$ | $\mathbb{Z}_2^2\oplus\mathbb{Z}_3$ | $\mathbb{Z}_2^2\oplus\mathbb{Z}_3$ | $\mathbb{Z}_1$ | $\mathbb{Z}_2^2\oplus\mathbb{Z}_3$ | $\mathbb{Z}\oplus\mathbb{Z}_2\oplus\mathbb{Z}_3$ | $\mathbb{Z}_2^2\oplus\mathbb{Z}_3^2$ | $\mathbb{Z}_2^2\oplus\mathbb{Z}_3$ | $\mathbb{Z}_2^2\oplus\mathbb{Z}_3$ |
| 151 | $\mathbb{Z}_1$ | $\mathbb{Z}_2^2$ | $\mathbb{Z}\oplus\mathbb{Z}_2$ | $\mathbb{Z}_2^2$ | $\mathbb{Z}_2^2$ | $\mathbb{Z}_2^2$ | $\mathbb{Z}_1$ | $\mathbb{Z}_2^2$ | $\mathbb{Z}\oplus\mathbb{Z}_2$ | $\mathbb{Z}_2^2$ | $\mathbb{Z}_2^2$ | $\mathbb{Z}_2^2$ |
| 152 | $\mathbb{Z}_1$ | $\mathbb{Z}_2^2\oplus\mathbb{Z}_3$ | $\mathbb{Z}\oplus\mathbb{Z}_2$ | $\mathbb{Z}_2^2$ | $\mathbb{Z}_2^2$ | $\mathbb{Z}_2^2$ | $\mathbb{Z}_1$ | $\mathbb{Z}_2^2\oplus\mathbb{Z}_3$ | $\mathbb{Z}\oplus\mathbb{Z}_2$ | $\mathbb{Z}_2^2$ | $\mathbb{Z}_2^2$ | $\mathbb{Z}_2^2$ |
| 153 | $\mathbb{Z}_1$ | $\mathbb{Z}_2^2$ | $\mathbb{Z}\oplus\mathbb{Z}_2$ | $\mathbb{Z}_2^2$ | $\mathbb{Z}_2^2$ | $\mathbb{Z}_2^2$ | $\mathbb{Z}_1$ | $\mathbb{Z}_2^2$ | $\mathbb{Z}\oplus\mathbb{Z}_2$ | $\mathbb{Z}_2^2$ | $\mathbb{Z}_2^2$ | $\mathbb{Z}_2^2$ |
| 154 | $\mathbb{Z}_1$ | $\mathbb{Z}_2^2\oplus\mathbb{Z}_3$ | $\mathbb{Z}\oplus\mathbb{Z}_2$ | $\mathbb{Z}_2^2$ | $\mathbb{Z}_2^2$ | $\mathbb{Z}_2^2$ | $\mathbb{Z}_1$ | $\mathbb{Z}_2^2\oplus\mathbb{Z}_3$ | $\mathbb{Z}\oplus\mathbb{Z}_2$ | $\mathbb{Z}_2^2$ | $\mathbb{Z}_2^2$ | $\mathbb{Z}_2^2$ |
| 155 | $\mathbb{Z}_1$ | $\mathbb{Z}_2^2$ | $\mathbb{Z}\oplus\mathbb{Z}_2$ | $\mathbb{Z}_2^2\oplus\mathbb{Z}_3$ | $\mathbb{Z}_2^2$ | $\mathbb{Z}_2^2$ | $\mathbb{Z}_1$ | $\mathbb{Z}_2^2$ | $\mathbb{Z}\oplus\mathbb{Z}_2$ | $\mathbb{Z}_2^2\oplus\mathbb{Z}_3$ | $\mathbb{Z}_2^2$ | $\mathbb{Z}_2^2$ |
| 156 | $\mathbb{Z}$ | $\mathbb{Z}_2$ | $\mathbb{Z}_2^2$ | $\mathbb{Z}_2^2\oplus\mathbb{Z}_3^3$ | $\mathbb{Z}_2^2\oplus\mathbb{Z}_3^3$ | $\mathbb{Z}_2^2$ | $\mathbb{Z}_2$ | $\mathbb{Z}\oplus\mathbb{Z}_2\oplus\mathbb{Z}_3^2$ | $\mathbb{Z}\oplus\mathbb{Z}_2\oplus\mathbb{Z}_3^2$ | $\mathbb{Z}_2^2$ | $\mathbb{Z}_2^2$ | $\mathbb{Z}_2^2\oplus\mathbb{Z}_3^3$ |
| 157 | $\mathbb{Z}$ | $\mathbb{Z}_2\oplus\mathbb{Z}_3$ | $\mathbb{Z}_2^2\oplus\mathbb{Z}_3$ | $\mathbb{Z}_2^2\oplus\mathbb{Z}_3^2$ | $\mathbb{Z}_2^2\oplus\mathbb{Z}_3^2$ | $\mathbb{Z}_2^2\oplus\mathbb{Z}_3$ | $\mathbb{Z}_2$ | $\mathbb{Z}\oplus\mathbb{Z}_2\oplus\mathbb{Z}_3$ | $\mathbb{Z}\oplus\mathbb{Z}_2\oplus\mathbb{Z}_3$ | $\mathbb{Z}_2^2\oplus\mathbb{Z}_3$ | $\mathbb{Z}_2^2\oplus\mathbb{Z}_3$ | $\mathbb{Z}_2^2\oplus\mathbb{Z}_3^2$ |
| 158 | $\mathbb{Z}$ | $\mathbb{Z}_1$ | $\mathbb{Z}_2$ | $\mathbb{Z}_3^3$ | $\mathbb{Z}_3^3$ | $\mathbb{Z}_1$ | $\mathbb{Z}_2$ | $\mathbb{Z}\oplus\mathbb{Z}_3^3$ | $\mathbb{Z}\oplus\mathbb{Z}_3^3$ | $\mathbb{Z}_1$ | $\mathbb{Z}_1$ | $\mathbb{Z}_3^3$ |
| 159 | $\mathbb{Z}$ | $\mathbb{Z}_3$ | $\mathbb{Z}_2\oplus\mathbb{Z}_3$ | $\mathbb{Z}_3^2$ | $\mathbb{Z}_3^2$ | $\mathbb{Z}_3$ | $\mathbb{Z}_2$ | $\mathbb{Z}\oplus\mathbb{Z}_3$ | $\mathbb{Z}\oplus\mathbb{Z}_3$ | $\mathbb{Z}_3$ | $\mathbb{Z}_3$ | $\mathbb{Z}_3^2$ |
| 160 | $\mathbb{Z}$ | $\mathbb{Z}_2$ | $\mathbb{Z}_2^2$ | $\mathbb{Z}_2^2\oplus\mathbb{Z}_3$ | $\mathbb{Z}_2^2\oplus\mathbb{Z}_3$ | $\mathbb{Z}_2^2$ | $\mathbb{Z}_2$ | $\mathbb{Z}\oplus\mathbb{Z}_2\oplus\mathbb{Z}_3$ | $\mathbb{Z}\oplus\mathbb{Z}_2$ | $\mathbb{Z}_2^2$ | $\mathbb{Z}_2^2$ | $\mathbb{Z}_2^2\oplus\mathbb{Z}_3$ |
| 161 | $\mathbb{Z}$ | $\mathbb{Z}_1$ | $\mathbb{Z}_2$ | $\mathbb{Z}_3$ | $\mathbb{Z}_3$ | $\mathbb{Z}_1$ | $\mathbb{Z}_2$ | $\mathbb{Z}\oplus\mathbb{Z}_3$ | $\mathbb{Z}$ | $\mathbb{Z}_1$ | $\mathbb{Z}_1$ | $\mathbb{Z}_3$ |
| 162 | $\mathbb{Z}_1$ | $\mathbb{Z}_2^3$ | $\mathbb{Z}_2^3\oplus\mathbb{Z}_3$ | $\mathbb{Z}_2^7\oplus\mathbb{Z}_3^2$ | $\mathbb{Z}_2^7$ | $\mathbb{Z}_2^{11}$ | $\mathbb{Z}_2$ | $\mathbb{Z}_2^2$ | $\mathbb{Z}\oplus\mathbb{Z}_2^4\oplus\mathbb{Z}_3$ | $\mathbb{Z}_2^5\oplus\mathbb{Z}_3$ | $\mathbb{Z}_2^9$ | $\mathbb{Z}_2^9$ |
| 163 | $\mathbb{Z}_1$ | $\mathbb{Z}_2^2$ | $\mathbb{Z}_2\oplus\mathbb{Z}_3$ | $\mathbb{Z}_2^3\oplus\mathbb{Z}_3^2$ | $\mathbb{Z}_2^2$ | $\mathbb{Z}_2^3$ | $\mathbb{Z}_2$ | $\mathbb{Z}_2^2$ | $\mathbb{Z}\oplus\mathbb{Z}_2^2\oplus\mathbb{Z}_3$ | $\mathbb{Z}_2\oplus\mathbb{Z}_3$ | $\mathbb{Z}_2^3$ | $\mathbb{Z}_2^2$ |
| 164 | $\mathbb{Z}_1$ | $\mathbb{Z}_2^3$ | $\mathbb{Z}_2^3$ | $\mathbb{Z}_2^7\oplus\mathbb{Z}_3^2$ | $\mathbb{Z}_2^7\oplus\mathbb{Z}_3$ | $\mathbb{Z}_2^{11}$ | $\mathbb{Z}_2$ | $\mathbb{Z}_2^2\oplus\mathbb{Z}_3$ | $\mathbb{Z}\oplus\mathbb{Z}_2^4\oplus\mathbb{Z}_3$ | $\mathbb{Z}_2^5$ | $\mathbb{Z}_2^9$ | $\mathbb{Z}_2^9\oplus\mathbb{Z}_3$ |
| 165 | $\mathbb{Z}_1$ | $\mathbb{Z}_2^2$ | $\mathbb{Z}_2$ | $\mathbb{Z}_2^3\oplus\mathbb{Z}_3^2$ | $\mathbb{Z}_2\oplus\mathbb{Z}_3$ | $\mathbb{Z}_2^3$ | $\mathbb{Z}_2$ | $\mathbb{Z}_2\oplus\mathbb{Z}_3$ | $\mathbb{Z}\oplus\mathbb{Z}_2^2\oplus\mathbb{Z}_3$ | $\mathbb{Z}_2$ | $\mathbb{Z}_2^3$ | $\mathbb{Z}_2\oplus\mathbb{Z}_3$ |
| 166 | $\mathbb{Z}_1$ | $\mathbb{Z}_2^3$ | $\mathbb{Z}_2^3$ | $\mathbb{Z}_2^7\oplus\mathbb{Z}_3$ | $\mathbb{Z}_2^7$ | $\mathbb{Z}_2^{11}$ | $\mathbb{Z}_2$ | $\mathbb{Z}_2^2$ | $\mathbb{Z}\oplus\mathbb{Z}_2^4$ | $\mathbb{Z}_2^5$ | $\mathbb{Z}_2^9$ | $\mathbb{Z}_2^9$ |
| 167 | $\mathbb{Z}_1$ | $\mathbb{Z}_2^2$ | $\mathbb{Z}_2$ | $\mathbb{Z}_2^3\oplus\mathbb{Z}_3$ | $\mathbb{Z}_2$ | $\mathbb{Z}_2^3$ | $\mathbb{Z}_2$ | $\mathbb{Z}_2$ | $\mathbb{Z}\oplus\mathbb{Z}_2^2$ | $\mathbb{Z}_2$ | $\mathbb{Z}_2^3$ | $\mathbb{Z}_2$ |
| 168 | $\mathbb{Z}$ | $\mathbb{Z}\oplus\mathbb{Z}_2\oplus\mathbb{Z}_3$ | $\mathbb{Z}\oplus\mathbb{Z}_2\oplus\mathbb{Z}_3$ | $\mathbb{Z}_2^2\oplus\mathbb{Z}_3^2$ | $\mathbb{Z}_2^2\oplus\mathbb{Z}_3^2$ | $\mathbb{Z}_2^2\oplus\mathbb{Z}_3^2$ | $\mathbb{Z}$ | $\mathbb{Z}\oplus\mathbb{Z}_2\oplus\mathbb{Z}_3$ | $\mathbb{Z}\oplus\mathbb{Z}_2\oplus\mathbb{Z}_3$ | $\mathbb{Z}_2^2\oplus\mathbb{Z}_3^2$ | $\mathbb{Z}_2^2\oplus\mathbb{Z}_3^2$ | $\mathbb{Z}_2^2\oplus\mathbb{Z}_3^2$ |
| 169 | $\mathbb{Z}$ | $\mathbb{Z}$ | $\mathbb{Z}$ | $\mathbb{Z}_1$ | $\mathbb{Z}_1$ | $\mathbb{Z}_1$ | $\mathbb{Z}$ | $\mathbb{Z}$ | $\mathbb{Z}$ | $\mathbb{Z}_1$ | $\mathbb{Z}_1$ | $\mathbb{Z}_1$ |
| 170 | $\mathbb{Z}$ | $\mathbb{Z}$ | $\mathbb{Z}$ | $\mathbb{Z}_1$ | $\mathbb{Z}_1$ | $\mathbb{Z}_1$ | $\mathbb{Z}$ | $\mathbb{Z}$ | $\mathbb{Z}$ | $\mathbb{Z}_1$ | $\mathbb{Z}_1$ | $\mathbb{Z}_1$ |
| 171 | $\mathbb{Z}$ | $\mathbb{Z}\oplus\mathbb{Z}_2$ | $\mathbb{Z}\oplus\mathbb{Z}_2$ | $\mathbb{Z}_2^2$ | $\mathbb{Z}_2^2$ | $\mathbb{Z}_2^2$ | $\mathbb{Z}$ | $\mathbb{Z}\oplus\mathbb{Z}_2$ | $\mathbb{Z}\oplus\mathbb{Z}_2$ | $\mathbb{Z}_2^2$ | $\mathbb{Z}_2^2$ | $\mathbb{Z}_2^2$ |
| 172 | $\mathbb{Z}$ | $\mathbb{Z}\oplus\mathbb{Z}_2$ | $\mathbb{Z}\oplus\mathbb{Z}_2$ | $\mathbb{Z}_2^2$ | $\mathbb{Z}_2^2$ | $\mathbb{Z}_2^2$ | $\mathbb{Z}$ | $\mathbb{Z}\oplus\mathbb{Z}_2$ | $\mathbb{Z}\oplus\mathbb{Z}_2$ | $\mathbb{Z}_2^2$ | $\mathbb{Z}_2^2$ | $\mathbb{Z}_2^2$ |
| 173 | $\mathbb{Z}$ | $\mathbb{Z}\oplus\mathbb{Z}_3$ | $\mathbb{Z}\oplus\mathbb{Z}_3$ | $\mathbb{Z}_3^2$ | $\mathbb{Z}_3^2$ | $\mathbb{Z}_3^2$ | $\mathbb{Z}$ | $\mathbb{Z}\oplus\mathbb{Z}_3$ | $\mathbb{Z}\oplus\mathbb{Z}_3$ | $\mathbb{Z}_3^2$ | $\mathbb{Z}_3^2$ | $\mathbb{Z}_3^2$ |
| 174 | $\mathbb{Z}_1$ | $\mathbb{Z}\oplus\mathbb{Z}_2^2\oplus\mathbb{Z}_3^2$ | $\mathbb{Z}_1$ | $\mathbb{Z}_2^4\oplus\mathbb{Z}_3^3$ | $\mathbb{Z}_1$ | $\mathbb{Z}_2^4\oplus\mathbb{Z}_3^3$ | $\mathbb{Z}\oplus\mathbb{Z}_2$ | $\mathbb{Z}_1$ | $\mathbb{Z}\oplus\mathbb{Z}_2^3\oplus\mathbb{Z}_3^2$ | $\mathbb{Z}_1$ | $\mathbb{Z}_2^4\oplus\mathbb{Z}_3^3$ | $\mathbb{Z}_1$ |
| 175 | $\mathbb{Z}_1$ | $\mathbb{Z}\oplus\mathbb{Z}_2^3\oplus\mathbb{Z}_3$ | $\mathbb{Z}_2^2$ | $\mathbb{Z}_2^8\oplus\mathbb{Z}_3^2$ | $\mathbb{Z}_2^6$ | $\mathbb{Z}_2^{12}\oplus\mathbb{Z}_3^2$ | $\mathbb{Z}\oplus\mathbb{Z}_2$ | $\mathbb{Z}_2$ | $\mathbb{Z}\oplus\mathbb{Z}_2^5\oplus\mathbb{Z}_3$ | $\mathbb{Z}_2^4$ | $\mathbb{Z}_2^{10}\oplus\mathbb{Z}_3^2$ | $\mathbb{Z}_2^8$ |
| 176 | $\mathbb{Z}_1$ | $\mathbb{Z}\oplus\mathbb{Z}_2^2\oplus\mathbb{Z}_3$ | $\mathbb{Z}_1$ | $\mathbb{Z}_2^4\oplus\mathbb{Z}_3^2$ | $\mathbb{Z}_1$ | $\mathbb{Z}_2^4\oplus\mathbb{Z}_3^2$ | $\mathbb{Z}\oplus\mathbb{Z}_2$ | $\mathbb{Z}_1$ | $\mathbb{Z}\oplus\mathbb{Z}_2^3\oplus\mathbb{Z}_3$ | $\mathbb{Z}_1$ | $\mathbb{Z}_2^4\oplus\mathbb{Z}_3^2$ | $\mathbb{Z}_1$ |
| 177 | $\mathbb{Z}_1$ | $\mathbb{Z}_2^3$ | $\mathbb{Z}\oplus\mathbb{Z}_2^3\oplus\mathbb{Z}_3$ | $\mathbb{Z}_2^6\oplus\mathbb{Z}_3^2$ | $\mathbb{Z}_2^8$ | $\mathbb{Z}_2^{10}$ | $\mathbb{Z}_1$ | $\mathbb{Z}_2^3$ | $\mathbb{Z}\oplus\mathbb{Z}_2^3\oplus\mathbb{Z}_3$ | $\mathbb{Z}_2^6\oplus\mathbb{Z}_3^2$ | $\mathbb{Z}_2^8$ | $\mathbb{Z}_2^{10}$ |
| 178 | $\mathbb{Z}_1$ | $\mathbb{Z}_2^2$ | $\mathbb{Z}\oplus\mathbb{Z}_2$ | $\mathbb{Z}_2^2$ | $\mathbb{Z}_2^2$ | $\mathbb{Z}_2^2$ | $\mathbb{Z}_1$ | $\mathbb{Z}_2^2$ | $\mathbb{Z}\oplus\mathbb{Z}_2$ | $\mathbb{Z}_2^2$ | $\mathbb{Z}_2^2$ | $\mathbb{Z}_2^2$ |
| 179 | $\mathbb{Z}_1$ | $\mathbb{Z}_2^2$ | $\mathbb{Z}\oplus\mathbb{Z}_2$ | $\mathbb{Z}_2^2$ | $\mathbb{Z}_2^2$ | $\mathbb{Z}_2^2$ | $\mathbb{Z}_1$ | $\mathbb{Z}_2^2$ | $\mathbb{Z}\oplus\mathbb{Z}_2$ | $\mathbb{Z}_2^2$ | $\mathbb{Z}_2^2$ | $\mathbb{Z}_2^2$ |
| 180 | $\mathbb{Z}_1$ | $\mathbb{Z}_2^3$ | $\mathbb{Z}\oplus\mathbb{Z}_2^3$ | $\mathbb{Z}_2^6$ | $\mathbb{Z}_2^8$ | $\mathbb{Z}_2^{10}$ | $\mathbb{Z}_1$ | $\mathbb{Z}_2^3$ | $\mathbb{Z}\oplus\mathbb{Z}_2^3$ | $\mathbb{Z}_2^6$ | $\mathbb{Z}_2^8$ | $\mathbb{Z}_2^{10}$ |
| 181 | $\mathbb{Z}_1$ | $\mathbb{Z}_2^3$ | $\mathbb{Z}\oplus\mathbb{Z}_2^3$ | $\mathbb{Z}_2^6$ | $\mathbb{Z}_2^8$ | $\mathbb{Z}_2^{10}$ | $\mathbb{Z}_1$ | $\mathbb{Z}_2^3$ | $\mathbb{Z}\oplus\mathbb{Z}_2^3$ | $\mathbb{Z}_2^6$ | $\mathbb{Z}_2^8$ | $\mathbb{Z}_2^{10}$ |
| 182 | $\mathbb{Z}_1$ | $\mathbb{Z}_2^2$ | $\mathbb{Z}\oplus\mathbb{Z}_2\oplus\mathbb{Z}_3$ | $\mathbb{Z}_2^2\oplus\mathbb{Z}_3^2$ | $\mathbb{Z}_2^2$ | $\mathbb{Z}_2^2$ | $\mathbb{Z}_1$ | $\mathbb{Z}_2^2$ | $\mathbb{Z}\oplus\mathbb{Z}_2\oplus\mathbb{Z}_3$ | $\mathbb{Z}_2^2\oplus\mathbb{Z}_3^2$ | $\mathbb{Z}_2^2$ | $\mathbb{Z}_2^2$ |
| 183 | $\mathbb{Z}$ | $\mathbb{Z}_2^2$ | $\mathbb{Z}_2^4$ | $\mathbb{Z}_2^6\oplus\mathbb{Z}_3^2$ | $\mathbb{Z}_2^8\oplus\mathbb{Z}_3^2$ | $\mathbb{Z}_2^{10}$ | $\mathbb{Z}_2$ | $\mathbb{Z}\oplus\mathbb{Z}_2^2\oplus\mathbb{Z}_3$ | $\mathbb{Z}\oplus\mathbb{Z}_2^3\oplus\mathbb{Z}_3$ | $\mathbb{Z}_2^6$ | $\mathbb{Z}_2^8$ | $\mathbb{Z}_2^{10}\oplus\mathbb{Z}_3^2$ |
| 184 | $\mathbb{Z}$ | $\mathbb{Z}_2$ | $\mathbb{Z}_2^2$ | $\mathbb{Z}_2^2\oplus\mathbb{Z}_3^2$ | $\mathbb{Z}_2^2\oplus\mathbb{Z}_3^2$ | $\mathbb{Z}_2^2$ | $\mathbb{Z}_2$ | $\mathbb{Z}\oplus\mathbb{Z}_2\oplus\mathbb{Z}_3$ | $\mathbb{Z}\oplus\mathbb{Z}_2\oplus\mathbb{Z}_3$ | $\mathbb{Z}_2^2$ | $\mathbb{Z}_2^2$ | $\mathbb{Z}_2^2\oplus\mathbb{Z}_3^2$ |
| 185 | $\mathbb{Z}$ | $\mathbb{Z}_2$ | $\mathbb{Z}_2^2$ | $\mathbb{Z}_2^2\oplus\mathbb{Z}_3^2$ | $\mathbb{Z}_2^2\oplus\mathbb{Z}_3^2$ | $\mathbb{Z}_2^2$ | $\mathbb{Z}_2$ | $\mathbb{Z}\oplus\mathbb{Z}_2\oplus\mathbb{Z}_3$ | $\mathbb{Z}\oplus\mathbb{Z}_2\oplus\mathbb{Z}_3$ | $\mathbb{Z}_2^2$ | $\mathbb{Z}_2^2$ | $\mathbb{Z}_2^2\oplus\mathbb{Z}_3^2$ |
| 186 | $\mathbb{Z}$ | $\mathbb{Z}_2$ | $\mathbb{Z}_2^2$ | $\mathbb{Z}_2^2\oplus\mathbb{Z}_3^2$ | $\mathbb{Z}_2^2\oplus\mathbb{Z}_3^2$ | $\mathbb{Z}_2^2$ | $\mathbb{Z}_2$ | $\mathbb{Z}\oplus\mathbb{Z}_2\oplus\mathbb{Z}_3$ | $\mathbb{Z}\oplus\mathbb{Z}_2\oplus\mathbb{Z}_3$ | $\mathbb{Z}_2^2$ | $\mathbb{Z}_2^2$ | $\mathbb{Z}_2^2\oplus\mathbb{Z}_3^2$ |

| No. | | | | | | | | | | | | |
|---|---|---|---|---|---|---|---|---|---|---|---|---|
| 187 | $\mathbb{Z}_1$ | $\mathbb{Z}_2^3$ | $\mathbb{Z}_2^3$ | $\mathbb{Z}_2^7\oplus\mathbb{Z}_3^3$ | $\mathbb{Z}_2^7$ | $\mathbb{Z}_2^{11}$ | $\mathbb{Z}_2$ | $\mathbb{Z}_2^2$ | $\mathbb{Z}\oplus\mathbb{Z}_2^4\oplus\mathbb{Z}_3^3$ | $\mathbb{Z}_2^5$ | $\mathbb{Z}_2^9$ | $\mathbb{Z}_2^9$ |
| 188 | $\mathbb{Z}_1$ | $\mathbb{Z}_2^2$ | $\mathbb{Z}_2$ | $\mathbb{Z}_2^3\oplus\mathbb{Z}_3^3$ | $\mathbb{Z}_2$ | $\mathbb{Z}_2^3$ | $\mathbb{Z}_2$ | $\mathbb{Z}_2$ | $\mathbb{Z}\oplus\mathbb{Z}_2^2\oplus\mathbb{Z}_3^3$ | $\mathbb{Z}_2$ | $\mathbb{Z}_2^3$ | $\mathbb{Z}_2$ |
| 189 | $\mathbb{Z}_1$ | $\mathbb{Z}_2^3\oplus\mathbb{Z}_3$ | $\mathbb{Z}_2^3$ | $\mathbb{Z}_2^7\oplus\mathbb{Z}_3^3$ | $\mathbb{Z}_2^7$ | $\mathbb{Z}_2^{11}\oplus\mathbb{Z}_3$ | $\mathbb{Z}_2$ | $\mathbb{Z}_2^2$ | $\mathbb{Z}\oplus\mathbb{Z}_2^4\oplus\mathbb{Z}_3$ | $\mathbb{Z}_2^5$ | $\mathbb{Z}_2^9\oplus\mathbb{Z}_3$ | $\mathbb{Z}_2^9$ |
| 190 | $\mathbb{Z}_1$ | $\mathbb{Z}_2^2\oplus\mathbb{Z}_3$ | $\mathbb{Z}_2$ | $\mathbb{Z}_2^3\oplus\mathbb{Z}_3^3$ | $\mathbb{Z}_2$ | $\mathbb{Z}_2^3\oplus\mathbb{Z}_3$ | $\mathbb{Z}_2$ | $\mathbb{Z}_2$ | $\mathbb{Z}\oplus\mathbb{Z}_2^2\oplus\mathbb{Z}_3$ | $\mathbb{Z}_2$ | $\mathbb{Z}_2^3\oplus\mathbb{Z}_3$ | $\mathbb{Z}_2$ |
| 191 | $\mathbb{Z}_1$ | $\mathbb{Z}_2^4$ | $\mathbb{Z}_2^6$ | $\mathbb{Z}_2^{14}\oplus\mathbb{Z}_3^2$ | $\mathbb{Z}_2^{20}$ | $\mathbb{Z}_2^{32}$ | $\mathbb{Z}_2$ | $\mathbb{Z}_2^3$ | $\mathbb{Z}\oplus\mathbb{Z}_2^7\oplus\mathbb{Z}_3$ | $\mathbb{Z}_2^{12}$ | $\mathbb{Z}_2^{22}$ | $\mathbb{Z}_2^{30}$ |
| 192 | $\mathbb{Z}_1$ | $\mathbb{Z}_2^3$ | $\mathbb{Z}_2^3$ | $\mathbb{Z}_2^7\oplus\mathbb{Z}_3^2$ | $\mathbb{Z}_2^7$ | $\mathbb{Z}_2^{11}$ | $\mathbb{Z}_2$ | $\mathbb{Z}_2^2$ | $\mathbb{Z}\oplus\mathbb{Z}_2^4\oplus\mathbb{Z}_3$ | $\mathbb{Z}_2^5$ | $\mathbb{Z}_2^9$ | $\mathbb{Z}_2^9$ |
| 193 | $\mathbb{Z}_1$ | $\mathbb{Z}_2^3$ | $\mathbb{Z}_2^3$ | $\mathbb{Z}_2^7\oplus\mathbb{Z}_3^2$ | $\mathbb{Z}_2^7$ | $\mathbb{Z}_2^{11}$ | $\mathbb{Z}_2$ | $\mathbb{Z}_2^2$ | $\mathbb{Z}\oplus\mathbb{Z}_2^4\oplus\mathbb{Z}_3$ | $\mathbb{Z}_2^5$ | $\mathbb{Z}_2^9$ | $\mathbb{Z}_2^9$ |
| 194 | $\mathbb{Z}_1$ | $\mathbb{Z}_2^3$ | $\mathbb{Z}_2^3$ | $\mathbb{Z}_2^7\oplus\mathbb{Z}_3^2$ | $\mathbb{Z}_2^7$ | $\mathbb{Z}_2^{11}$ | $\mathbb{Z}_2$ | $\mathbb{Z}_2^2$ | $\mathbb{Z}\oplus\mathbb{Z}_2^4\oplus\mathbb{Z}_3$ | $\mathbb{Z}_2^5$ | $\mathbb{Z}_2^9$ | $\mathbb{Z}_2^9$ |
| 195 | $\mathbb{Z}_1$ | $\mathbb{Z}_2\oplus\mathbb{Z}_3$ | $\mathbb{Z}\oplus\mathbb{Z}_2^3$ | $\mathbb{Z}_2^4\oplus\mathbb{Z}_3$ | $\mathbb{Z}_2^4\oplus\mathbb{Z}_3$ | $\mathbb{Z}_2^8\oplus\mathbb{Z}_3$ | $\mathbb{Z}_1$ | $\mathbb{Z}_2\oplus\mathbb{Z}_3$ | $\mathbb{Z}\oplus\mathbb{Z}_2^3$ | $\mathbb{Z}_2^4\oplus\mathbb{Z}_3$ | $\mathbb{Z}_2^4\oplus\mathbb{Z}_3$ | $\mathbb{Z}_2^8\oplus\mathbb{Z}_3$ |
| 196 | $\mathbb{Z}_1$ | $\mathbb{Z}_3$ | $\mathbb{Z}\oplus\mathbb{Z}_2^2\oplus\mathbb{Z}_4$ | $\mathbb{Z}_2^2\oplus\mathbb{Z}_3$ | $\mathbb{Z}_3$ | $\mathbb{Z}_2^6\oplus\mathbb{Z}_3$ | $\mathbb{Z}_1$ | $\mathbb{Z}_3$ | $\mathbb{Z}\oplus\mathbb{Z}_2^2\oplus\mathbb{Z}_4$ | $\mathbb{Z}_2^2\oplus\mathbb{Z}_3$ | $\mathbb{Z}_3$ | $\mathbb{Z}_2^6\oplus\mathbb{Z}_3$ |
| 197 | $\mathbb{Z}_1$ | $\mathbb{Z}_4\oplus\mathbb{Z}_3$ | $\mathbb{Z}\oplus\mathbb{Z}_2$ | $\mathbb{Z}_2^2\oplus\mathbb{Z}_3$ | $\mathbb{Z}_2^2\oplus\mathbb{Z}_3$ | $\mathbb{Z}_2^4\oplus\mathbb{Z}_3$ | $\mathbb{Z}_1$ | $\mathbb{Z}_4\oplus\mathbb{Z}_3$ | $\mathbb{Z}\oplus\mathbb{Z}_2$ | $\mathbb{Z}_2^2\oplus\mathbb{Z}_3$ | $\mathbb{Z}_2^2\oplus\mathbb{Z}_3$ | $\mathbb{Z}_2^4\oplus\mathbb{Z}_3$ |
| 198 | $\mathbb{Z}_1$ | $\mathbb{Z}_3$ | $\mathbb{Z}$ | $\mathbb{Z}_3$ | $\mathbb{Z}_3$ | $\mathbb{Z}_3$ | $\mathbb{Z}_1$ | $\mathbb{Z}_3$ | $\mathbb{Z}$ | $\mathbb{Z}_3$ | $\mathbb{Z}_3$ | $\mathbb{Z}_3$ |
| 199 | $\mathbb{Z}_1$ | $\mathbb{Z}_2\oplus\mathbb{Z}_3$ | $\mathbb{Z}$ | $\mathbb{Z}_2\oplus\mathbb{Z}_3$ | $\mathbb{Z}_2\oplus\mathbb{Z}_3$ | $\mathbb{Z}_2\oplus\mathbb{Z}_3$ | $\mathbb{Z}_1$ | $\mathbb{Z}_2\oplus\mathbb{Z}_3$ | $\mathbb{Z}$ | $\mathbb{Z}_2\oplus\mathbb{Z}_3$ | $\mathbb{Z}_2\oplus\mathbb{Z}_3$ | $\mathbb{Z}_2\oplus\mathbb{Z}_3$ |
| 200 | $\mathbb{Z}_1$ | $\mathbb{Z}_2^2\oplus\mathbb{Z}_3$ | $\mathbb{Z}_2^4$ | $\mathbb{Z}_2^{10}\oplus\mathbb{Z}_3$ | $\mathbb{Z}_2^{12}$ | $\mathbb{Z}_2^{22}\oplus\mathbb{Z}_3$ | $\mathbb{Z}_2$ | $\mathbb{Z}_2$ | $\mathbb{Z}\oplus\mathbb{Z}_2^5$ | $\mathbb{Z}_2^8$ | $\mathbb{Z}_2^{14}\oplus\mathbb{Z}_3$ | $\mathbb{Z}_2^{20}$ |
| 201 | $\mathbb{Z}_1$ | $\mathbb{Z}_2^2\oplus\mathbb{Z}_3$ | $\mathbb{Z}_2\oplus\mathbb{Z}_4$ | $\mathbb{Z}_2^4\oplus\mathbb{Z}_3$ | $\mathbb{Z}_2^2$ | $\mathbb{Z}_2^6\oplus\mathbb{Z}_3$ | $\mathbb{Z}_2$ | $\mathbb{Z}_2$ | $\mathbb{Z}\oplus\mathbb{Z}_2^3$ | $\mathbb{Z}_2^2$ | $\mathbb{Z}_2^4\oplus\mathbb{Z}_3$ | $\mathbb{Z}_2^4$ |
| 202 | $\mathbb{Z}_1$ | $\mathbb{Z}_2\oplus\mathbb{Z}_3$ | $\mathbb{Z}_2^3$ | $\mathbb{Z}_2^6\oplus\mathbb{Z}_3$ | $\mathbb{Z}_2^4$ | $\mathbb{Z}_2^{11}\oplus\mathbb{Z}_3$ | $\mathbb{Z}_2$ | $\mathbb{Z}_1$ | $\mathbb{Z}\oplus\mathbb{Z}_2^4$ | $\mathbb{Z}_2^4$ | $\mathbb{Z}_2^6\oplus\mathbb{Z}_3$ | $\mathbb{Z}_2^9$ |
| 203 | $\mathbb{Z}_1$ | $\mathbb{Z}_2\oplus\mathbb{Z}_3$ | $\mathbb{Z}_2\oplus\mathbb{Z}_4$ | $\mathbb{Z}_2^3\oplus\mathbb{Z}_3$ | $\mathbb{Z}_1$ | $\mathbb{Z}_2^5\oplus\mathbb{Z}_3$ | $\mathbb{Z}_2$ | $\mathbb{Z}_1$ | $\mathbb{Z}\oplus\mathbb{Z}_2^3$ | $\mathbb{Z}_2$ | $\mathbb{Z}_2^2\oplus\mathbb{Z}_3$ | $\mathbb{Z}_2^3$ |
| 204 | $\mathbb{Z}_1$ | $\mathbb{Z}_2^2\oplus\mathbb{Z}_3$ | $\mathbb{Z}_2^2$ | $\mathbb{Z}_2^6\oplus\mathbb{Z}_3$ | $\mathbb{Z}_2^6$ | $\mathbb{Z}_2^{12}\oplus\mathbb{Z}_3$ | $\mathbb{Z}_2$ | $\mathbb{Z}_4$ | $\mathbb{Z}\oplus\mathbb{Z}_2^3$ | $\mathbb{Z}_2^4$ | $\mathbb{Z}_2^8\oplus\mathbb{Z}_3$ | $\mathbb{Z}_2^{10}$ |
| 205 | $\mathbb{Z}_1$ | $\mathbb{Z}_2\oplus\mathbb{Z}_3$ | $\mathbb{Z}_1$ | $\mathbb{Z}_2^2\oplus\mathbb{Z}_3$ | $\mathbb{Z}_1$ | $\mathbb{Z}_2^2\oplus\mathbb{Z}_3$ | $\mathbb{Z}_2$ | $\mathbb{Z}_1$ | $\mathbb{Z}\oplus\mathbb{Z}_2$ | $\mathbb{Z}_1$ | $\mathbb{Z}_2^2\oplus\mathbb{Z}_3$ | $\mathbb{Z}_1$ |
| 206 | $\mathbb{Z}_1$ | $\mathbb{Z}_2^2\oplus\mathbb{Z}_3$ | $\mathbb{Z}_2$ | $\mathbb{Z}_2^3\oplus\mathbb{Z}_3$ | $\mathbb{Z}_2$ | $\mathbb{Z}_2^3\oplus\mathbb{Z}_3$ | $\mathbb{Z}_2$ | $\mathbb{Z}_2$ | $\mathbb{Z}\oplus\mathbb{Z}_2^2$ | $\mathbb{Z}_2$ | $\mathbb{Z}_2^3\oplus\mathbb{Z}_3$ | $\mathbb{Z}_2$ |
| 207 | $\mathbb{Z}_1$ | $\mathbb{Z}_2^2$ | $\mathbb{Z}\oplus\mathbb{Z}_2^2\oplus\mathbb{Z}_4$ | $\mathbb{Z}_2^3\oplus\mathbb{Z}_4^2\oplus\mathbb{Z}_3$ | $\mathbb{Z}_2^6$ | $\mathbb{Z}_2^9$ | $\mathbb{Z}_1$ | $\mathbb{Z}_2^2$ | $\mathbb{Z}\oplus\mathbb{Z}_2^2\oplus\mathbb{Z}_4$ | $\mathbb{Z}_2^3\oplus\mathbb{Z}_4^2\oplus\mathbb{Z}_3$ | $\mathbb{Z}_2^6$ | $\mathbb{Z}_2^9$ |
| 208 | $\mathbb{Z}_1$ | $\mathbb{Z}_2^2$ | $\mathbb{Z}\oplus\mathbb{Z}_2^3$ | $\mathbb{Z}_2^5\oplus\mathbb{Z}_3$ | $\mathbb{Z}_2^6$ | $\mathbb{Z}_2^9$ | $\mathbb{Z}_1$ | $\mathbb{Z}_2^2$ | $\mathbb{Z}\oplus\mathbb{Z}_2^3$ | $\mathbb{Z}_2^5\oplus\mathbb{Z}_3$ | $\mathbb{Z}_2^6$ | $\mathbb{Z}_2^9$ |
| 209 | $\mathbb{Z}_1$ | $\mathbb{Z}_2$ | $\mathbb{Z}\oplus\mathbb{Z}_2^3$ | $\mathbb{Z}_2^3\oplus\mathbb{Z}_4\oplus\mathbb{Z}_3$ | $\mathbb{Z}_2^4$ | $\mathbb{Z}_2^8$ | $\mathbb{Z}_1$ | $\mathbb{Z}_2$ | $\mathbb{Z}\oplus\mathbb{Z}_2^3$ | $\mathbb{Z}_2^3\oplus\mathbb{Z}_4\oplus\mathbb{Z}_3$ | $\mathbb{Z}_2^4$ | $\mathbb{Z}_2^8$ |
| 210 | $\mathbb{Z}_1$ | $\mathbb{Z}_2$ | $\mathbb{Z}\oplus\mathbb{Z}_2^2$ | $\mathbb{Z}_2^2\oplus\mathbb{Z}_3$ | $\mathbb{Z}_2$ | $\mathbb{Z}_2^4$ | $\mathbb{Z}_1$ | $\mathbb{Z}_2$ | $\mathbb{Z}\oplus\mathbb{Z}_2^2$ | $\mathbb{Z}_2^2\oplus\mathbb{Z}_3$ | $\mathbb{Z}_2$ | $\mathbb{Z}_2^4$ |
| 211 | $\mathbb{Z}_1$ | $\mathbb{Z}_2^2$ | $\mathbb{Z}\oplus\mathbb{Z}_2^2$ | $\mathbb{Z}_2^3\oplus\mathbb{Z}_4\oplus\mathbb{Z}_3$ | $\mathbb{Z}_2^5$ | $\mathbb{Z}_2^7$ | $\mathbb{Z}_1$ | $\mathbb{Z}_2^2$ | $\mathbb{Z}\oplus\mathbb{Z}_2^2$ | $\mathbb{Z}_2^3\oplus\mathbb{Z}_4\oplus\mathbb{Z}_3$ | $\mathbb{Z}_2^5$ | $\mathbb{Z}_2^7$ |
| 212 | $\mathbb{Z}_1$ | $\mathbb{Z}_2$ | $\mathbb{Z}$ | $\mathbb{Z}_2\oplus\mathbb{Z}_3$ | $\mathbb{Z}_2$ | $\mathbb{Z}_2$ | $\mathbb{Z}_1$ | $\mathbb{Z}_2$ | $\mathbb{Z}$ | $\mathbb{Z}_2\oplus\mathbb{Z}_3$ | $\mathbb{Z}_2$ | $\mathbb{Z}_2$ |
| 213 | $\mathbb{Z}_1$ | $\mathbb{Z}_2$ | $\mathbb{Z}$ | $\mathbb{Z}_2\oplus\mathbb{Z}_3$ | $\mathbb{Z}_2$ | $\mathbb{Z}_2$ | $\mathbb{Z}_1$ | $\mathbb{Z}_2$ | $\mathbb{Z}$ | $\mathbb{Z}_2\oplus\mathbb{Z}_3$ | $\mathbb{Z}_2$ | $\mathbb{Z}_2$ |
| 214 | $\mathbb{Z}_1$ | $\mathbb{Z}_2^2$ | $\mathbb{Z}\oplus\mathbb{Z}_2$ | $\mathbb{Z}_2^3\oplus\mathbb{Z}_3$ | $\mathbb{Z}_2^4$ | $\mathbb{Z}_2^5$ | $\mathbb{Z}_1$ | $\mathbb{Z}_2^2$ | $\mathbb{Z}\oplus\mathbb{Z}_2$ | $\mathbb{Z}_2^3\oplus\mathbb{Z}_3$ | $\mathbb{Z}_2^4$ | $\mathbb{Z}_2^5$ |
| 215 | $\mathbb{Z}_1$ | $\mathbb{Z}_2^2$ | $\mathbb{Z}_2^3$ | $\mathbb{Z}_2^4\oplus\mathbb{Z}_4^2\oplus\mathbb{Z}_3$ | $\mathbb{Z}_2^5\oplus\mathbb{Z}_3$ | $\mathbb{Z}_2^{10}$ | $\mathbb{Z}_2$ | $\mathbb{Z}_2\oplus\mathbb{Z}_3$ | $\mathbb{Z}\oplus\mathbb{Z}_2^3\oplus\mathbb{Z}_4$ | $\mathbb{Z}_2^4$ | $\mathbb{Z}_2^7$ | $\mathbb{Z}_2^8\oplus\mathbb{Z}_3$ |
| 216 | $\mathbb{Z}_1$ | $\mathbb{Z}_2$ | $\mathbb{Z}_2^3$ | $\mathbb{Z}_2^3\oplus\mathbb{Z}_4^2\oplus\mathbb{Z}_3$ | $\mathbb{Z}_2^5\oplus\mathbb{Z}_3$ | $\mathbb{Z}_2^9$ | $\mathbb{Z}_2$ | $\mathbb{Z}_3$ | $\mathbb{Z}\oplus\mathbb{Z}_2^3\oplus\mathbb{Z}_4$ | $\mathbb{Z}_2^3$ | $\mathbb{Z}_2^5$ | $\mathbb{Z}_2^7\oplus\mathbb{Z}_3$ |
| 217 | $\mathbb{Z}_1$ | $\mathbb{Z}_2\oplus\mathbb{Z}_4$ | $\mathbb{Z}_2^2$ | $\mathbb{Z}_2^2\oplus\mathbb{Z}_4^2\oplus\mathbb{Z}_3$ | $\mathbb{Z}_2^5\oplus\mathbb{Z}_3$ | $\mathbb{Z}_2^5\oplus\mathbb{Z}_4$ | $\mathbb{Z}_2$ | $\mathbb{Z}_2\oplus\mathbb{Z}_3$ | $\mathbb{Z}\oplus\mathbb{Z}_2^2$ | $\mathbb{Z}_2^2$ | $\mathbb{Z}_2^5$ | $\mathbb{Z}_2^4\oplus\mathbb{Z}_3$ |
| 218 | $\mathbb{Z}_1$ | $\mathbb{Z}_4$ | $\mathbb{Z}_2^2$ | $\mathbb{Z}_2\oplus\mathbb{Z}_4^2\oplus\mathbb{Z}_3$ | $\mathbb{Z}_2^2\oplus\mathbb{Z}_3$ | $\mathbb{Z}_2^3\oplus\mathbb{Z}_4^2$ | $\mathbb{Z}_2$ | $\mathbb{Z}_3$ | $\mathbb{Z}\oplus\mathbb{Z}_2^3$ | $\mathbb{Z}_2$ | $\mathbb{Z}_2^4$ | $\mathbb{Z}_2^3\oplus\mathbb{Z}_3$ |
| 219 | $\mathbb{Z}_1$ | $\mathbb{Z}_2$ | $\mathbb{Z}_2^2$ | $\mathbb{Z}_4^2\oplus\mathbb{Z}_3$ | $\mathbb{Z}_3$ | $\mathbb{Z}_2^2\oplus\mathbb{Z}_4^2$ | $\mathbb{Z}_2$ | $\mathbb{Z}_3$ | $\mathbb{Z}\oplus\mathbb{Z}_2^2\oplus\mathbb{Z}_4$ | $\mathbb{Z}_1$ | $\mathbb{Z}_2^2$ | $\mathbb{Z}_2^2\oplus\mathbb{Z}_3$ |
| 220 | $\mathbb{Z}_1$ | $\mathbb{Z}_4$ | $\mathbb{Z}_1$ | $\mathbb{Z}_2\oplus\mathbb{Z}_4\oplus\mathbb{Z}_3$ | $\mathbb{Z}_3$ | $\mathbb{Z}_2\oplus\mathbb{Z}_4$ | $\mathbb{Z}_2$ | $\mathbb{Z}_3$ | $\mathbb{Z}\oplus\mathbb{Z}_2$ | $\mathbb{Z}_1$ | $\mathbb{Z}_2\oplus\mathbb{Z}_4$ | $\mathbb{Z}_3$ |
| 221 | $\mathbb{Z}_1$ | $\mathbb{Z}_2^3$ | $\mathbb{Z}_2^5$ | $\mathbb{Z}_2^{10}\oplus\mathbb{Z}_4^2\oplus\mathbb{Z}_3$ | $\mathbb{Z}_2^{16}$ | $\mathbb{Z}_2^{27}$ | $\mathbb{Z}_2$ | $\mathbb{Z}_2^2$ | $\mathbb{Z}\oplus\mathbb{Z}_2^5\oplus\mathbb{Z}_4$ | $\mathbb{Z}_2^{10}$ | $\mathbb{Z}_2^{18}$ | $\mathbb{Z}_2^{25}$ |
| 222 | $\mathbb{Z}_1$ | $\mathbb{Z}_2^2$ | $\mathbb{Z}_2\oplus\mathbb{Z}_4$ | $\mathbb{Z}_2^2\oplus\mathbb{Z}_4^2\oplus\mathbb{Z}_3$ | $\mathbb{Z}_2^3$ | $\mathbb{Z}_2^5\oplus\mathbb{Z}_4$ | $\mathbb{Z}_2$ | $\mathbb{Z}_2$ | $\mathbb{Z}\oplus\mathbb{Z}_2^3$ | $\mathbb{Z}_2\oplus\mathbb{Z}_4$ | $\mathbb{Z}_2^5$ | $\mathbb{Z}_2^4$ |
| 223 | $\mathbb{Z}_1$ | $\mathbb{Z}_2^2$ | $\mathbb{Z}_2^3$ | $\mathbb{Z}_2^5\oplus\mathbb{Z}_4^2\oplus\mathbb{Z}_3$ | $\mathbb{Z}_2^8$ | $\mathbb{Z}_2^{14}$ | $\mathbb{Z}_2$ | $\mathbb{Z}_2$ | $\mathbb{Z}\oplus\mathbb{Z}_2^4$ | $\mathbb{Z}_2^5$ | $\mathbb{Z}_2^{10}$ | $\mathbb{Z}_2^{12}$ |
| 224 | $\mathbb{Z}_1$ | $\mathbb{Z}_2^3$ | $\mathbb{Z}_2^4$ | $\mathbb{Z}_2^7\oplus\mathbb{Z}_4\oplus\mathbb{Z}_3$ | $\mathbb{Z}_2^8$ | $\mathbb{Z}_2^{13}$ | $\mathbb{Z}_2$ | $\mathbb{Z}_2^2$ | $\mathbb{Z}\oplus\mathbb{Z}_2^5$ | $\mathbb{Z}_2^6$ | $\mathbb{Z}_2^{10}$ | $\mathbb{Z}_2^{11}$ |
| 225 | $\mathbb{Z}_1$ | $\mathbb{Z}_2^2$ | $\mathbb{Z}_2^4$ | $\mathbb{Z}_2^7\oplus\mathbb{Z}_4^2\oplus\mathbb{Z}_3$ | $\mathbb{Z}_2^{11}$ | $\mathbb{Z}_2^{20}$ | $\mathbb{Z}_2$ | $\mathbb{Z}_2$ | $\mathbb{Z}\oplus\mathbb{Z}_2^5$ | $\mathbb{Z}_2^7$ | $\mathbb{Z}_2^{13}$ | $\mathbb{Z}_2^{18}$ |
| 226 | $\mathbb{Z}_1$ | $\mathbb{Z}_2^2$ | $\mathbb{Z}_2^3$ | $\mathbb{Z}_2^4\oplus\mathbb{Z}_4^2\oplus\mathbb{Z}_3$ | $\mathbb{Z}_2^5$ | $\mathbb{Z}_2^{10}$ | $\mathbb{Z}_2$ | $\mathbb{Z}_2$ | $\mathbb{Z}\oplus\mathbb{Z}_2^3\oplus\mathbb{Z}_4$ | $\mathbb{Z}_2^4$ | $\mathbb{Z}_2^7$ | $\mathbb{Z}_2^8$ |
| 227 | $\mathbb{Z}_1$ | $\mathbb{Z}_2^2$ | $\mathbb{Z}_2^3$ | $\mathbb{Z}_2^5\oplus\mathbb{Z}_4\oplus\mathbb{Z}_3$ | $\mathbb{Z}_2^5$ | $\mathbb{Z}_2^{10}$ | $\mathbb{Z}_2$ | $\mathbb{Z}_2$ | $\mathbb{Z}\oplus\mathbb{Z}_2^4$ | $\mathbb{Z}_2^4$ | $\mathbb{Z}_2^7$ | $\mathbb{Z}_2^8$ |
| 228 | $\mathbb{Z}_1$ | $\mathbb{Z}_2^2$ | $\mathbb{Z}_2^2$ | $\mathbb{Z}_2^2\oplus\mathbb{Z}_4\oplus\mathbb{Z}_3$ | $\mathbb{Z}_2$ | $\mathbb{Z}_2^3\oplus\mathbb{Z}_4$ | $\mathbb{Z}_2$ | $\mathbb{Z}_2$ | $\mathbb{Z}\oplus\mathbb{Z}_2^3$ | $\mathbb{Z}_2$ | $\mathbb{Z}_2^3$ | $\mathbb{Z}_2^2$ |
| 229 | $\mathbb{Z}_1$ | $\mathbb{Z}_2^3$ | $\mathbb{Z}_2^4$ | $\mathbb{Z}_2^7\oplus\mathbb{Z}_4^2\oplus\mathbb{Z}_3$ | $\mathbb{Z}_2^{11}$ | $\mathbb{Z}_2^{18}$ | $\mathbb{Z}_2$ | $\mathbb{Z}_2^2$ | $\mathbb{Z}\oplus\mathbb{Z}_2^5$ | $\mathbb{Z}_2^7$ | $\mathbb{Z}_2^{13}$ | $\mathbb{Z}_2^{16}$ |
| 230 | $\mathbb{Z}_1$ | $\mathbb{Z}_2^2$ | $\mathbb{Z}_2$ | $\mathbb{Z}_2^2\oplus\mathbb{Z}_4\oplus\mathbb{Z}_3$ | $\mathbb{Z}_2^2$ | $\mathbb{Z}_2^3\oplus\mathbb{Z}_4$ | $\mathbb{Z}_2$ | $\mathbb{Z}_2$ | $\mathbb{Z}\oplus\mathbb{Z}_2^2$ | $\mathbb{Z}_2$ | $\mathbb{Z}_2^4$ | $\mathbb{Z}_2^2$ |

**Appendix D: Nontrivial differential $d_2^{1,2}$ in the standard LHS spectral sequence for No. 42 ($Fmm2$)**

In this appendix, we perform the calculation

$$d_2^{1,2}: H^1(P, H^2(T, \mathbb{Z}_2)) \to H^3(P, H^1(T, \mathbb{Z}_2)) \tag{96}$$

for group $G = $ No. 42 ($Fmm2$), which gives

$$d_2(\omega_{12}) = (A_c^3 + A_c A_m^2)\omega_{01}. \tag{97}$$

This group is a symmorphic group whose corresponding extension class $H_\rho^2(P, T)$ is trivial (see Table III). Yet, following [43, 129], we prove that $d_2^{1,2}$ is still nontrivial because of the nontrivial action of $P$ on $T$.

First let us write down an explicit cochain expression $\omega_{12} \in C^1(P, C^2(T, \mathbb{Z}_2))$ which represents the nontrivial element of $H^1(P, H^2(T, \mathbb{Z}_2)) \cong \mathbb{Z}_2$. Note that $H^2(T, \mathbb{Z}_2) \cong (\mathbb{Z}_2)^3$, and is generated by three elements whose cochain representatives can be chosen to be

$$b_1(t_1, t_2) = y_1 z_2, \quad b_2(t_1, t_2) = x_1 z_2, \quad b_3(t_1, t_2) = x_1 y_2, \tag{98}$$

for group elements $t_i = T_1^{x_i} T_2^{y_i} T_3^{z_i} \in T, i = 1, 2$. To consider the action of $P$ on $b_{1,2,3}$, first we write down its action on the generators $T_{1,2,3}$ of $T$,

$$C_2: \ T_1 \to T_2 T_3^{-1}, \quad T_2 \to T_1 T_3^{-1}, \quad T_3 \to T_3^{-1}, \tag{99a}$$

$$M: \ T_1 \to T_2 T_3^{-1}, \quad T_2 \to T_2, \quad T_3 \to T_1^{-1} T_2, \tag{99b}$$

$$C_2 M: \ T_1 \to T_1, \quad T_2 \to T_1 T_3^{-1}, \quad T_3 \to T_1 T_2^{-1}. \tag{99c}$$

Then its action on $b_{1,2,3}$ is dictated by

$$(C_2.b)(t_1, t_2) = b(C_2^{-1} t_1 C_2, C_2^{-1} t_2 C_2) = b(T_1^{y_1} T_2^{x_1} T_3^{-x_1 - y_1 - z_1}, T_1^{y_2} T_2^{x_2} T_3^{-x_2 - y_2 - z_2}), \tag{100a}$$

$$(M.b)(t_1, t_2) = b(M^{-1} t_1 M, M^{-1} t_2 M) = b(T_1^{-z_1} T_2^{x_1 + y_1 + z_1} T_3^{-x_1}, T_1^{-z_2} T_2^{x_2 + y_2 + z_2} T_3^{-x_2}), \tag{100b}$$

$$((C_2 M).b)(t_1, t_2) = b(M^{-1} C_2^{-1} t_1 C_2 M, M^{-1} C_2^{-1} t_2 C_2 M) = b(T_1^{x_1 + y_1 + z_1} T_2^{-z_1} T_3^{-y_1}, T_1^{x_2 + y_2 + z_2} T_2^{-z_2} T_3^{-y_2}). \tag{100c}$$

Thus, the elements of $P$ map $[b_{1,2,3}] \in H^2(T, \mathbb{Z}_2)$ in the following way,

$$C_2: \ [b_1] \to [b_2 + b_3], \quad [b_2] \to [b_1 + b_3], \quad [b_3] \to [b_3], \tag{101a}$$

$$M: \ [b_1] \to [b_2 + b_3], \quad [b_2] \to [b_2], \quad [b_3] \to [b_1 + b_2], \tag{101b}$$

$$C_2 M: \ [b_1] \to [b_1], \quad [b_2] \to [b_1 + b_3], \quad [b_3] \to [b_1 + b_2], \tag{101c}$$

For $\omega_{12}: P \to H^2(T, \mathbb{Z}_2)$, define

$$(\delta_P \omega_{12})(p_1, p_2) := p_1.(\omega_{12}(p_2)) + \omega_{12}(p_1) - \omega_{12}(p_1 p_2), \quad p_1, p_2 \in P. \tag{102}$$

we should have $[(\delta_P \omega_{12})(p_1, p_2)]$ equals 0 in $H^2(T, \mathbb{Z}_2)$. Accordingly, we can choose the cochain representative $\omega_{12}$ to be

$$\omega_{12}: \ C_2 \mapsto b_3, \quad M \mapsto b_2, \quad C_2 M \mapsto b_1. \tag{103}$$

The nontriviality of this cochain representative can be seen by restricting to the subgroup generated by, e.g., $C_2$, which remains nontrivial in the subgroup.

To obtain the image of the differential of LHS spectral sequence, we attempt to promote $\omega_{12}$ into a cocycle in $H^3(G, \mathbb{Z}_2)$, following the ansatz outlined in [43]. In particular, for the given $\omega_{12} \in C^1(P, C^2(T, \mathbb{Z}_2))$, we attempt to solve the following sets of equations for some $f_{21} \in C^2(P, C^1(T, \mathbb{Z}_2))$ [21],

$$\delta_T \omega_{12} = 0, \tag{104a}$$

$$\delta_P \omega_{12} + \delta_T f_{21} = 0, \tag{104b}$$

$$\delta_P f_{21} = 0. \tag{104c}$$

Eq. (104a) is automatically satisfied for our choice of $\omega_{12}$. However, $(\delta_P \omega_{12})(p_1, p_2)$ may not be a zero cochain in $C^2(T, \mathbb{Z}_2)$, and we have to find a nontrivial $f_{21}$ that satisfies Eq. (104b). This fact will contribute to $d_2^{1,2}$. Now we explicitly perform the calculation of the cochains $(\delta_P \omega_{12})(p_1, p_2)$ using Eq. (102) to demonstrate this fact:

$$\begin{aligned}
((\delta_P \omega_{12})(C_2, C_2))(t_1, t_2) &= y_1 x_2 + x_1 y_2, \\
((\delta_P \omega_{12})(M, M))(t_1, t_2) &= z_1 x_2 + x_1 z_2, \\
((\delta_P \omega_{12})(C_2 M, C_2 M))(t_1, t_2) &= z_1 y_2 + y_1 z_2, \\
((\delta_P \omega_{12})(C_2, M))(t_1, t_2) &= y_1(x_2 + y_2 + z_2) + x_1 y_2 + y_1 z_2, \\
((\delta_P \omega_{12})(M, C_2))(t_1, t_2) &= z_1(x_2 + y_2 + z_2) + x_1 z_2 + y_1 z_2, \\
((\delta_P \omega_{12})(C_2, C_2 M))(t_1, t_2) &= x_1(x_2 + y_2 + z_2) + x_1 y_2 + x_1 z_2, \\
((\delta_P \omega_{12})(C_2 M, C_2))(t_1, t_2) &= (x_1 + y_1 + z_1) z_2 + y_1 z_2 + x_1 z_2, \\
((\delta_P \omega_{12})(M, C_2 M))(t_1, t_2) &= (x_1 + y_1 + z_1) x_2 + x_1 z_2 + x_1 y_2, \\
((\delta_P \omega_{12})(C_2 M, M))(t_1, t_2) &= (x_1 + y_1 + z_1) y_2 + y_1 z_2 + x_1 y_2,
\end{aligned} \tag{105}$$

we can check that

$$(\delta_P \omega_{12})(p_1, p_2)(t_1, t_2) = (\delta_T(f_{21}(p_1, p_2)))(t_1, t_2), \quad p_1, p_2 \in P, \ t_1, t_2 \in T, \tag{106}$$

———————

where we defined $f_{21}(id, p) = f_{21}(p, id) = 0$ for $p \in P$ and

[21] These are equations (B83), (B84) and (B85) in [43], and we simplify it for the special group structure and module that we are considering.

$$f_{21}(C_2, C_2)(t) = xy, \qquad f_{21}(M, M)(t) = xz, \qquad f_{21}(C_2M, C_2M)(t) = yz,$$

$$f_{21}(C_2, M)(t) = \frac{y(y+1)}{2} + xy, \qquad f_{21}(M, C_2)(t) = \frac{z(z+1)}{2} + xz + yz, \qquad f_{21}(C_2, C_2M)(t) = \frac{x(x+1)}{2}, \tag{107}$$

$$f_{21}(C_2M, M)(t) = \frac{y(y+1)}{2} + yz, \quad f_{21}(M, C_2M)(t) = \frac{x(x+1)}{2} + xy + xz, \quad f_{21}(C_2M, C_2)(t) = \frac{z(z+1)}{2}.$$

However, the obtained $f_{21} \in C^2(P, C^1(T, \mathbb{Z}_2))$ does not satisfy Eq. (104c). We continue to define

$$f_{31}(p_1, p_2, p_3)(t) := (\delta_P f_{21})(p_1, p_2, p_3)(t) = [p_1.f_{21}(p_2, p_3) + f_{21}(p_1 p_2, p_3) + f_{21}(p_1, p_2 p_3) + f_{21}(p_1, p_2)](t).$$

$f_{31}$ is a cochain in $C^3(P, C^1(T, \mathbb{Z}_2))$. Moreover, one can explicitly check using elementary methods that $[f_{31}]: P \times P \times P \to H^1(T, \mathbb{Z}_2)$ is a representative cocycle for a nontrivial element in $E_2^{3,1} = H^3(P, H^1(T, \mathbb{Z}_2))$. Here we list all the nonzero maps of $f_{31}$:

$$
\begin{array}{lll}
f_{31}(M, C_2, M) = A_z, & f_{31}(M, C_2M, M) = A_x, & f_{31}(M, C_2, C_2) = A_z, \\
f_{31}(M, C_2, C_2M) = A_z, & f_{31}(M, C_2M, C_2) = A_x, & f_{31}(M, C_2M, C_2M) = A_x, \\
f_{31}(C_2, M, M) = A_y, & f_{31}(C_2M, M, M) = A_y, & f_{31}(C_2, M, C_2) = A_z, \\
f_{31}(C_2, M, C_2M) = A_y, & f_{31}(C_2M, M, C_2M) = A_{y+z}, & f_{31}(C_2, C_2M, M) = A_x, \\
f_{31}(C_2M, C_2M, M) = A_z, & f_{31}(C_2, C_2M, C_2) = A_{x+y+z}, & f_{31}(C_2, C_2M, C_2M) = A_x, \\
f_{31}(C_2M, C_2, C_2) = A_z, & f_{31}(C_2M, C_2M, C_2) = A_y. &
\end{array}
\tag{108}
$$

$f_{31}$ is exactly the obstruction of promoting $\omega_{12}$ to a cocycle in $H^3(G, \mathbb{Z}_2)$, and from the explicit cochain representative in Eq. (108), we can check that it indeed represents $(A_c^3 + A_c A_m^2)\omega_{01}$. To summarize, we have the nonzero differential

$$d_2^{1,2}: \omega_{12} \mapsto (A_c^3 + A_c A_m^2)\omega_{01}. \tag{109}$$

This nonzero differential serves as an example for the statement of Corollary 1 of Ref. [44].

## Appendix E: High-degree relations in the mod-2 cohomology ring of space groups No. 219, 228, and 230

In this appendix, we outline the calculation of high-degree ($n \geq 7$) relations of the mod-2 cohomology rings for space groups No. 219, 228, and 230. Because of the memory cost, we are unable to get the relations directly from our program. We did the calculation by restricting to the subgroups not containing the three-fold rotation and, because the restriction map is injective as proven in Theorem 12, it suffices to check the relations in the subgroup. A fourth group, No. 226, also has high-degree ($n \geq 7$) relations, but we are unable to obtain them, leaving No. 226 the only space group whose complete expressions of the associated mod-2 cohomology ring is missing.

### A: No. 219 ($F\bar{4}3c$)

The mod-2 cohomology ring of No. 219 contains two degree-6 generators, $F_\gamma$ and $F_\delta$, and relations at degree $n \geq 7$. These high-degree relations are obtained by restricting to the subgroup No. 120. The restriction map of cohomology generators can be obtained in GAP

$$A_m|_{219} \to (A_m)|_{120}, \tag{110a}$$

$$B_\alpha|_{219} \to (B_\alpha + A_{c'}^2)|_{120}, \tag{110b}$$

$$B_{xy+xz+yz}|_{219} \to (B_{z(x+y)} + A_{c'}A_{x+y+z})|_{120}, \tag{110c}$$

$$C_\alpha|_{219} \to (A_{c'}B_\alpha)|_{120}, \tag{110d}$$

$$C_\beta|_{219} \to (A_{x+y+z}B_\alpha)|_{120}, \tag{110e}$$

$$F_\gamma|_{219} \to \left(B_\alpha(D_\gamma + B_\alpha^2 + B_\alpha B_{z(x+y)} + B_\alpha A_{c'}A_{x+y+z} + A_{c'}^4)\right)\Big|_{120}, \tag{110f}$$

$$F_\delta|_{219} \to \left(B_\alpha(D_\delta + A_{c'}^3 A_{x+y+z})\right)\Big|_{120}. \tag{110g}$$

Using the relations of No. 120 it is easy to see that

$$(F_\gamma A_m)|_{219} = (F_\delta A_m)|_{219} = 0, \tag{111}$$

$$(F_\gamma B_{xy+xz+yz})|_{219} \to \left[(B_{z(x+y)} + A_{c'}A_{x+y+z})B_\alpha(D_\gamma + B_\alpha^2 + B_\alpha B_{z(x+y)} + B_\alpha A_{c'}A_{x+y+z} + A_{c'}^4)\right]|_{120}$$
$$= \left[(B_\alpha^2 + A_{c'}^2)(B_\alpha(D_\delta + A_{c'}^3 A_{x+y+z}) + B_\alpha^2 A_{c'}A_{x+y+z})\right]|_{120} \tag{112}$$
$$\to (B_\alpha(F_\delta + C_\alpha C_\beta))|_{219},$$

$$(F_\delta B_{xy+xz+yz})|_{219} \to \left[(B_{z(x+y)} + A_{c'}A_{x+y+z})B_\alpha(D_\delta + A_{c'}^3 A_{x+y+z})\right]|_{120}$$
$$= \left[(B_\alpha + A_{c'}^2)B_\alpha(D_\delta + A_{c'}^3 A_{x+y+z})\right]|_{120} \tag{113}$$
$$\to (B_\alpha F_\delta)|_{219},$$

$$(F_\gamma C_\beta)|_{219} \to \left[A_{x+y+z}B_\alpha^2(D_\gamma + B_\alpha^2 + B_\alpha B_{z(x+y)} + B_\alpha A_{c'}A_{x+y+z} + A_{c'}^4)\right]|_{120}$$
$$= \left[A_{x+y+z}B_\alpha^2(D_\delta + B_\alpha A_{c'}^2 + A_{c'}^3 A_{x+y+z})\right]|_{120} \tag{114}$$
$$\to (C_\beta F_\delta + C_\alpha^2 C_\beta)|_{219},$$

$$(F_\delta C_\beta)|_{219} \to \left[A_{x+y+z}B_\alpha^2(D_\delta + A_{c'}^3 A_{x+y+z})\right]|_{120}$$
$$= \left[A_{c'}B_\alpha^2(D_\delta + A_{c'}^3 A_{x+y+z})\right]|_{120} \tag{115}$$
$$\to (C_\alpha F_\delta)|_{219},$$

$$(F_\gamma^2)|_{219} \to \left[B_\alpha^2(D_\gamma^2 + B_\alpha^4 + B_\alpha^2 B_{z(x+y)}^2 + B_\alpha^2 A_{c'}^2 A_{x+y+z}^2 + A_{c'}^8)\right]\bigg|_{120}$$
$$= \left[B_\alpha^2 A_{c'}^2\left(B_\alpha(D_\gamma + B_\alpha^2 + B_\alpha B_{z(x+y)} + B_\alpha A_{c'}A_{x+y+z} + A_{c'}^4) + (B_\alpha + A_{c'}^2)^3 + A_{c'}^2 B_\alpha^2\right)\right]\bigg|_{120} \tag{116}$$
$$\to \left(C_\alpha^2(F_\gamma + B_\alpha^3 + C_\alpha^2)\right)|_{219},$$

$$(F_\delta^2)|_{219} \to \left[B_\alpha^2(D_\delta^2 + A_{c'}^6 A_{x+y+z}^2)\right]\bigg|_{120}$$
$$= \left[B_\alpha^2 A_{c'}A_{x+y+z}(B_\alpha(D_\delta + A_{c'}^3 A_{x+y+z}) + (B_{z(x+y)} + A_{c'}A_{x+y+z})^3 + B_\alpha^2 A_{c'}A_{x+y+z})\right]|_{120} \tag{117}$$
$$\to \left(C_\alpha C_\beta(F_\delta + B_{xy+xz+yz}^3 + C_\alpha C_\beta)\right)|_{219},$$

and finally,

$$(F_\gamma F_\delta)|_{219} \to \left[B_\alpha^2(D_\gamma + B_\alpha^2 + B_\alpha B_{z(x+y)} + B_\alpha A_{c'}A_{x+y+z} + A_{c'}^4)(D_\delta + A_{c'}^3 A_{x+y+z})\right]\bigg|_{120}$$
$$= \left[B_\alpha^2 A_{c'}A_{x+y+z}(B_\alpha^3 + B_\alpha A_{c'}^4 + A_{c'}^6)\right]\bigg|_{120} \tag{118}$$
$$\to \left(C_\alpha C_\beta(B_\alpha^3 + C_\alpha^2)\right)|_{219}.$$

### B: No. 228 ($Fd\bar{3}c$)

The mod-2 cohomology ring of No. 228 contains a degree-6 generator, $F_\gamma$, and relations at degree $n \geq 7$. We determine the high-degree relations via restricting to subgroup No. 142. However, unlike No. 219 and No. 230, GAP cannot be used to directly output the restriction of $F_\gamma$ in No. 142 due to the high memory cost to construct the resolution up to degree 7. Hence, we have to use another approach. The generator $F_\gamma$ can be uniquely determined (as a cocycle vector in GAP) by computing the integral cohomology (see the analysis below), and the ambiguity of the restriction can be resolved by examining the mod-2 reduction of integral cohomology with a different $\mathbb{Z}$ module for distinct $G$ actions. This restriction then allows us to write down the high-degree ($n \geq 7$) relations for the mod-2 cohomology ring of No. 228.

First, we calculate the image of $F_\gamma$ under restriction explicitly. The group generators of No. 142 embed into group No. 228 in the following way

$$(T_1)|_{142} \to (T_1)|_{228},$$
$$(T_2)|_{142} \to (T_2 T_3^{-1})|_{228},$$
$$(T_3)|_{142} \to (T_1^{-1} T_3)|_{228},$$
$$(C_2)|_{142} \to (C_2)|_{228},$$
$$(C_2')|_{142} \to (T_1^{-1} T_3 C_2 GI)|_{228}, \tag{119}$$
$$(G)|_{142} \to (C_2' I)|_{228},$$
$$(I)|_{142} \to (I)|_{228}.$$

From this, we have

$$
\begin{aligned}
(A_i)|_{228} &\to (A_i + A_{c'} + A_m)|_{142}, \\
(A_m)|_{228} &\to (A_{c'})|_{142}, \\
(B_\alpha)|_{228} &\to (A_i^2 + A_i A_{c'} + B_\alpha)|_{142}, \\
(C_\alpha)|_{228} &\to (A_m B_\alpha)|_{142}.
\end{aligned}
\tag{120}
$$

Note that for the (untwisted) integral cohomology, we have $H^6(\text{No. } 228, \mathbb{Z}) \cong \mathbb{Z}_2^3 \oplus \mathbb{Z}_4$. Their mod-2 reduction in $H^6(\text{No. } 228, \mathbb{Z}_2)$ is generated by

$$
\langle A_i^6, A_i^4 A_m^2, C_\alpha^2; F_\gamma \rangle|_{228},
\tag{121}
$$

with the last entry the degree-6 generator of the mod-2 cohomology ring, defined by the mod-2 reduction of the $\mathbb{Z}_4$ summand [22]. The mod-2 reduction of $H^6(\text{No. } 142, \mathbb{Z}_2) \cong \mathbb{Z}_2^4 \oplus \mathbb{Z}_4$ is generated by

$$
\langle A_{c'}^6, A_i^6, A_i^5 A_m, A_i^2 B_\alpha^2; B_\alpha^3 + B_\alpha D_\gamma \rangle|_{142},
\tag{122}
$$

again with the last entry the mod-2 reduction of the $\mathbb{Z}_4$ summand. Therefore, the mod-2 reduction of $F_\gamma$ must have the form

$$
(B_\alpha^3 + B_\alpha D_\gamma + x_1 A_{c'}^6 + x_2 A_i^6 + x_3 A_i^5 A_m + x_4 A_i^2 B_\alpha^2)|_{142}, \quad x_{1,2,3,4} \in \{0,1\}.
\tag{123}
$$

To resolve the ambiguity, consider the twisted module $\mathbb{Z}^I$, where the superscript denotes (all) the generators of the group that act nontrivially on the $\mathbb{Z}$ module (the nontrivial action is $n \mapsto -n, n \in \mathbb{Z}$). From Eq. (119) it is clear that the module $\mathbb{Z}^I$ for No. 228 corresponds to the module $\mathbb{Z}^{C_2', G, I}$ for No. 142. We have $H^6(\text{No. } 228, \mathbb{Z}^I) \cong \mathbb{Z}_2^2 \oplus \mathbb{Z}_4$ and $H^6(\text{No. } 142, \mathbb{Z}^{C_2', G, I}) = \mathbb{Z}_2^3 \oplus \mathbb{Z}_4$. A direct check using GAP determines the generating sets of their mod-2 reduction:

$$
\langle A_i^5 A_m + A_i^4 A_m^2, C_\alpha^2; F_\gamma + A_i^5 A_m \rangle|_{228},
\tag{124}
$$

and

$$
\langle A_{c'}^5 A_i, A_i^5 A_m, A_i^6 + A_i^2 B_\alpha^2; A_i^6 + B_\alpha^3 + B_\alpha D_\gamma \rangle|_{142}.
\tag{125}
$$

Therefore, we can write the mod-2 reduction of $(F_\gamma + A_i^5 A_m)|_{228}$ to be

$$
(B_\alpha^3 + B_\alpha D_\gamma + A_i^6 + y_1 A_{c'}^5 A_i + y_2 A_i^5 A_m + y_3(A_i^6 + A_i^2 B_\alpha^2))|_{142}, \quad y_{1,2,3} \in \{0,1\}.
\tag{126}
$$

Combining Eq. (123), Eq. (126) and $(A_i^5 A_m)|_{228} \to (A_{c'}^6 + A_{c'}^5 A_i)|_{142}$, we must have

$$
x_1 = 1, \quad y_1 = 1, \quad x_2 = y_3 + 1, \quad x_3 = y_2, \quad x_4 = y_3,
\tag{127}
$$

implying that the allowed restriction has the form

$$
F_\gamma|_{228} \to \left(B_\alpha D_\gamma + B_\alpha^3 + A_i^6 + A_{c'}^6 + y_2 A_i^5 A_m + y_3(A_i^6 + A_i^2 B_\alpha^2)\right)|_{142}.
\tag{128}
$$

To obtain the high-degree relations for No. 228, we need to compute the multiplication of $F_\gamma$ by $A_i, A_m, B_\alpha, C_\alpha$, and $F_\gamma$ itself (some of them may not give rise to a relation):

$$
\begin{aligned}
(A_m F_\gamma)|_{228} &\to \left(A_{c'}(B_\alpha D_\gamma + B_\alpha^3 + A_i^6 + A_{c'}^6 + y_2 A_i^5 A_m + y_3(A_i^6 + A_i^2 B_\alpha^2))\right)|_{142} \\
&= \left(A_{c'}(B_\alpha^3 + A_{c'}^6)\right)|_{142} \\
&= \left(A_{c'}^7 + A_{c'}(A_i^2 + B_\alpha)^3\right)|_{142} \\
&\to \left(A_m^7 + A_m B_\alpha^3\right)|_{228} \\
&= A_m(A_i^6 + B_\alpha^3)|_{228},
\end{aligned}
\tag{129}
$$

where we have used the relations $(A_{c'} A_i^2)|_{142} = 0$ and $(A_{c'} D_\gamma)|_{142} = 0$;

$$
\begin{aligned}
(A_i F_\gamma)|_{228} &\to \left((A_i + A_m + A_{c'})(B_\alpha D_\gamma + B_\alpha^3 + A_i^6 + A_{c'}^6 + y_2 A_i^5 A_m + y_3(A_i^6 + A_i^2 B_\alpha^2))\right)|_{142} \\
&= \left((A_i + A_m + A_{c'})(B_\alpha^3 + A_i^6 + A_{c'}^6)\right)|_{142} \\
&= \left(A_i^2(A_i + A_m)B_\alpha^2 + (A_i + A_m)A_i^6 + (A_i + A_{c'})A_{c'}^6\right)|_{142} \\
&= \left((A_i + A_m)(A_i^2 B_\alpha^2 + A_i^6 + A_i^5 A_m) + A_{c'}^2(A_{c'}^5 + A_{c'}^4 A_i)\right)|_{142} \\
&\to \left((A_i + A_m)C_\alpha^2 + A_m^2 A_i^5\right)|_{228} \\
&= \left(A_m^2 A_i^5\right)|_{228},
\end{aligned}
\tag{130}
$$

———————

[22] We can explicit check, based on the result of $H^6(\text{No. } 228, \mathbb{Z})$ and its mod-2 reduction, that the mod-2 reduction of the $\mathbb{Z}_4$ summand is not given by cohomology generators of lower degrees.

where we have used the relations $(A_{c'}A_m)|_{142} = 0$, $(A_m(A_i + A_m))|_{142} = 0$, $(A_{c'}A_i^2)|_{142} = 0$, $((A_i + A_m)(A_i^2 + B_\alpha) + A_{c'}B_\alpha)|_{142} = 0$, $(A_i(A_i + A_m)(A_i^2 + B_\alpha))|_{142} = 0$, $(A_{c'}D_\gamma)|_{142} = 0$, and $(A_iC_\alpha)|_{228} = (A_mC_\alpha)|_{228} = 0$.

Note that neither does $(B_\alpha F_\gamma)|_{228}$ nor $(C_\alpha F_\gamma)|_{228}$ lead to a relation.

Finally, we solve for the relation at degree 12. The existence of such a relation in the group No. 228 is guaranteed by the degree-12 relation $\left(F_\gamma^2 + C_\alpha^2(F_\gamma + B_\alpha^3 + C_\alpha^2)\right)|_{219} = 0$ in the group No. 219: the degree 12 relation of No. 228 restricts to this relation in No. 219. Below we solve for the relation satisfied by $(F_\gamma^2)|_{228}$. We will see this also enforces that $y_2 = y_3$.

We have

$$
\begin{aligned}
(F_\gamma^2)|_{228} \to & \left((A_i^{12} + A_{c'}^{12} + B_\alpha^6 + B_\alpha^2 D_\gamma^2 + y_2 A_i^{10} A_m^2 + y_3(A_i^{12} + A_i^4 B_\alpha^4))\right)|_{142} \\
= & \left(A_{c'}^{12} + (A_i^2 + B_\alpha)^6 + (A_m B_\alpha)^4 + (A_m B_\alpha)^2(A_i^6 + A_{c'}^6 + B_\alpha^3 + B_\alpha D_\gamma) + y_2 A_i^{10} A_m^2 + y_3(A_i^{12} + A_i^4 B_\alpha^4)\right)|_{142},
\end{aligned}
$$
(131)

where the intermediate steps are (below we used the relation $\left(A_i^8 + A_i^7 A_m + A_i^2 B_\alpha^3 + A_i^2 B_\alpha D_\gamma + D_\gamma^2\right)|_{142} = 0$)

$$
\begin{aligned}
& \left((A_i^{12} + A_{c'}^{12} + B_\alpha^6 + B_\alpha^2 D_\gamma^2)\right)|_{142} \\
= & \left((A_i^{12} + A_{c'}^{12} + B_\alpha^6 + B_\alpha^2(A_i^8 + A_i^7 A_m + A_i^2(B_\alpha^3 + B_\alpha D_\gamma)))\right)|_{142} \\
= & \left(A_i^{12} + A_{c'}^{12} + B_\alpha^6 + B_\alpha^2 A_i^8 + B_\alpha^2 A_i^7 A_m + (B_\alpha^2(A_i^2 B_\alpha^3 + A_i^3(A_i + A_m)B_\alpha^2 + A_i A_m B_\alpha^3)) + B_\alpha^2 A_i A_m(A_{c'}^6 + B_\alpha D_\gamma))\right)|_{142} \\
= & \left(A_i^{12} + A_{c'}^{12} + B_\alpha^6 + A_i^8 B_\alpha^2 + B_\alpha^4 A_i^3(A_i + A_m)) + B_\alpha^2 A_i A_m(A_i^6 + A_{c'}^6 + B_\alpha^3 + B_\alpha D_\gamma))\right)|_{142} \\
= & \left(A_{c'}^{12} + A_i^{12} + A_i^8 B_\alpha^2 + A_i^4 B_\alpha^4 + B_\alpha^6 + A_m^4 B_\alpha^4 + B_\alpha^2 A_m^2(A_i^6 + A_{c'}^6 + B_\alpha^3 + B_\alpha D_\gamma))\right)|_{142} \\
= & \left(A_{c'}^{12} + (A_i^2 + B_\alpha)^6 + (A_m B_\alpha)^4 + (A_m B_\alpha)^2(A_i^6 + A_{c'}^6 + B_\alpha^3 + B_\alpha D_\gamma))\right)|_{142}.
\end{aligned}
$$
(132)

one can examine whether there exists a degree-12 polynomial in No. 228 that restricts to the last line of Eq. (131). It turns out that when $y_2 \neq y_3$ no such polynomial can be found. On the other hand, when $y_2 = y_3 = y$, one can explicitly check that the last line of Eq. (131)

$$
\to \left(A_m^{12} + B_\alpha^6 + C_\alpha^4 + C_\alpha^2(F_\gamma + yC_\alpha^2) + yC_\alpha^4\right)|_{228} = \left(A_m^{12} + B_\alpha^6 + C_\alpha^4 + C_\alpha^2 F_\gamma\right)|_{228},
$$

therefore we must have $y_2 = y_3 = y = 0$ or 1, and for either case the degree-12 relation is

$$
\left(F_\gamma^2 + C_\alpha^2 F_\gamma + C_\alpha^4 + B_\alpha^6 + A_m^{12}\right)|_{228} = 0.
$$
(133)

### C: No. 230 ($Ia\bar{3}d$)

The mod-2 cohomology ring of No. 230 contains a degree-4 generator, $D_\gamma$, and a high-degree relation at degree $n = 8$. This relation is obtained using restriction to subgroup No. 142. GAP gives the restriction map to No. 142:

$$
(A_i)|_{230} \to (A_i)|_{142}, \tag{134a}
$$
$$
(A_m)|_{230} \to (A_m)|_{142}, \tag{134b}
$$
$$
(B_\alpha)|_{230} \to \left(A_i^2 + A_i A_{c'} + A_{c'}^2 + A_m^2 + B_\alpha\right)|_{142}. \tag{134c}
$$

Note that we have $((A_m + A_i)A_i B_\alpha)|_{230} \to \left((A_m + A_i)A_i(A_i^2 + A_i A_{c'} + A_{c'}^2 + A_m^2 + B_\alpha)\right)|_{142} = 0$, as expected.

We define $(D_\gamma)|_{230}$ such that it restricts to $(D_\gamma)|_{142}$:

$$
(D_\gamma)|_{230} \to (D_\gamma)|_{142}. \tag{135}
$$

The vector-represented expression $(D_\gamma)|_{230}$ can be obtained in GAP. We have $(A_i^2 B_\alpha^3)|_{230} \to \left(A_i^8 + A_i^7 A_m + A_i^2 B_\alpha^3\right)|_{142}$, $(A_i^2 B_\alpha D_\gamma)|_{230} \to (A_i^2 B_\alpha D_\gamma)|_{142}$. Therefore the degree-8 relation for No. 230 is

$$
(A_i^2 B_\alpha^3 + A_i^2 B_\alpha D_\gamma + D_\gamma^2)|_{230} = 0. \tag{136}
$$

### Appendix F: Collection of results for space groups No. 1–230

This appendix contains the main results of this paper: a complete list of the mod-2 cohomology rings for space groups No. 1–230 (except for No. 226 whose high-degree ($n \geq 7$) relations are incomplete); and for each space group $G$, its IWPs, their associated cohomology element in $H^3(G, \mathbb{Z}_2)$, and their associated topological invariants.

For completeness, we also give the group generators as coordinate transformations, where the coordinate setup agrees with the "Standard/Default Setting" on Bilbao Crystallographic Server [31] or International Tables for Crystallography (ITC) [32].

**No. 1:** $P1$

This group is generated by three translations $T_{1,2,3}$ as given in Eqs. (88). The $\mathbb{Z}_2$ cohomology ring is given by

$$\mathbb{Z}_2[A_x, A_y, A_z]/\langle \mathcal{R}_2 \rangle \tag{137}$$

where the relations are

$$\mathcal{R}_2: \quad A_x^2, \quad A_y^2, \quad A_z^2. \tag{138a}$$

We have the following table regarding IWPs and group cohomology at degree 3.

| Wyckoff position | Little group Intl. | Little group Schönflies | Coordinates | LSM anomaly class | Topo. inv. |
|---|---|---|---|---|---|
| 1a | 1 | $C_1$ | $(x, y, z)$ | $A_x A_y A_z$ | $\varphi_3[T_1, T_2, T_3]$ |

**No. 2:** $P\bar{1}$

This group is generated by three translations $T_{1,2,3}$ as given in Eqs. (88), and an inversion $I$:

$$I: (x, y, z) \rightarrow (-x, -y, -z). \tag{139a}$$

The $\mathbb{Z}_2$ cohomology ring is given by

$$\mathbb{Z}_2[A_i, A_x, A_y, A_z]/\langle \mathcal{R}_2 \rangle \tag{140}$$

where the relations are

$$\mathcal{R}_2: \quad A_x(A_i + A_x), \quad A_y(A_i + A_y), \quad A_z(A_i + A_z). \tag{141a}$$

We have the following table regarding IWPs and group cohomology at degree 3.

| Wyckoff position | Little group Intl. | Little group Schönflies | Coordinates | LSM anomaly class | Topo. inv. |
|---|---|---|---|---|---|
| 1a | $\bar{1}$ | $C_i$ | $(0, 0, 0)$ | $(A_i + A_x)(A_i + A_y)(A_i + A_z)$ | $\varphi_1[I]$ |
| 1b | $\bar{1}$ | $C_i$ | $(0, 0, 1/2)$ | $(A_i + A_x)(A_i + A_y)A_z$ | $\varphi_1[T_3 I]$ |
| 1c | $\bar{1}$ | $C_i$ | $(0, 1/2, 0)$ | $(A_i + A_x)A_y(A_i + A_z)$ | $\varphi_1[T_2 I]$ |
| 1d | $\bar{1}$ | $C_i$ | $(1/2, 0, 0)$ | $A_x(A_i + A_y)(A_i + A_z)$ | $\varphi_1[T_1 I]$ |
| 1e | $\bar{1}$ | $C_i$ | $(1/2, 1/2, 0)$ | $A_x A_y(A_i + A_z)$ | $\varphi_1[T_1 T_2 I]$ |
| 1f | $\bar{1}$ | $C_i$ | $(1/2, 0, 1/2)$ | $A_x(A_i + A_y)A_z$ | $\varphi_1[T_1 T_3 I]$ |
| 1g | $\bar{1}$ | $C_i$ | $(0, 1/2, 1/2)$ | $(A_i + A_x)A_y A_z$ | $\varphi_1[T_2 T_3 I]$ |
| 1h | $\bar{1}$ | $C_i$ | $(1/2, 1/2, 1/2)$ | $A_x A_y A_z$ | $\varphi_1[T_1 T_2 T_3 I]$ |

**No. 3:** $P2$

This group is generated by three translations $T_{1,2,3}$ as given in Eqs. (88), and a two-fold rotation $C_2$:

$$C_2: (x, y, z) \rightarrow (-x, y, -z). \tag{142a}$$

The $\mathbb{Z}_2$ cohomology ring is given by

$$\mathbb{Z}_2[A_c, A_x, A_y, A_z]/\langle \mathcal{R}_2 \rangle \tag{143}$$

where the relations are

$$\mathcal{R}_2: \quad A_x(A_c + A_x), \quad A_y^2, \quad A_z(A_c + A_z). \tag{144a}$$

We have the following table regarding IWPs and group cohomology at degree 3.

| Wyckoff position | Little group Intl. | Little group Schönflies | Coordinates | LSM anomaly class | Topo. inv. |
|---|---|---|---|---|---|
| 1a | 2 | $C_2$ | $(0, y, 0)$ | $(A_c + A_x)A_y(A_c + A_z)$ | $\varphi_2[T_2, C_2]$ |
| 1b | 2 | $C_2$ | $(0, y, 1/2)$ | $(A_c + A_x)A_y A_z$ | $\varphi_2[T_2, T_3 C_2]$ |
| 1c | 2 | $C_2$ | $(1/2, y, 0)$ | $A_x A_y(A_c + A_z)$ | $\varphi_2[T_2, T_1 C_2]$ |
| 1d | 2 | $C_2$ | $(1/2, y, 1/2)$ | $A_x A_y A_z$ | $\varphi_2[T_2, T_1 T_3 C_2]$ |

**No. 4:** $P2_1$

This group is generated by three translations $T_{1,2,3}$ as given in Eqs. (88), and a two-fold screw $S_2$:

$$S_2\colon (x,y,z) \to (-x, y+1/2, -z). \tag{145a}$$

The $\mathbb{Z}_2$ cohomology ring is given by

$$\mathbb{Z}_2[A_c, A_x, A_z]/\langle \mathcal{R}_2 \rangle \tag{146}$$

where the relations are

$$\mathcal{R}_2\colon \quad A_c^2, \quad A_x(A_c + A_x), \quad A_z(A_c + A_z). \tag{147a}$$

We have the following table regarding IWPs and group cohomology at degree 3.

| Wyckoff position | Little group Intl. | Little group Schönflies | Coordinates | LSM anomaly class | Topo. inv. |
|---|---|---|---|---|---|
| 2a | 1 | $C_1$ | $(x,y,z), (-x, y+1/2, -z)$ | $A_c A_x A_z$ | $\widehat{\varphi_3}[T_1, T_3, S_2]$ |

Here the topological invariant can be chosen to be

$$\begin{aligned}\widehat{\varphi_3}[T_1, T_3, S_2] =& \lambda(T_1, T_3, T_1^{-1}T_3^{-1}S_2) + \lambda(T_3, T_1, T_1^{-1}T_3^{-1}S_2) + \lambda(T_1, T_1^{-1}S_2, T_3) + \lambda(T_3, T_3^{-1}S_2, T_1) \\ &+ \lambda(S_2, T_1, T_3) + \lambda(S_2, T_3, T_1).\end{aligned} \tag{148}$$

**No. 5:** $C2$

This group is generated by three translations $T_{1,2,3}$ as given in Eqs. (89), and a two-fold rotation $C_2$:

$$C_2\colon (x,y,z) \to (-x, y, -z). \tag{149a}$$

The $\mathbb{Z}_2$ cohomology ring is given by

$$\mathbb{Z}_2[A_c, A_{x+y}, A_z, B_{xy}]/\langle \mathcal{R}_2, \mathcal{R}_3, \mathcal{R}_4 \rangle \tag{150}$$

where the relations are

$$\mathcal{R}_2\colon \quad A_c A_{x+y}, \quad A_{x+y}^2, \quad A_z(A_c + A_z), \tag{151a}$$
$$\mathcal{R}_3\colon \quad A_{x+y}B_{xy}, \tag{151b}$$
$$\mathcal{R}_4\colon \quad B_{xy}^2. \tag{151c}$$

We have the following table regarding IWPs and group cohomology at degree 3.

| Wyckoff position | Little group Intl. | Little group Schönflies | Coordinates $(0,0,0)+(1/2,1/2,0)+$ | LSM anomaly class | Topo. inv. |
|---|---|---|---|---|---|
| 2a | 2 | $C_2$ | $(0,y,0)$ | $(A_c + A_z)B_{xy}$ | $\varphi_2[T_1 T_2, C_2]$ |
| 2b | 2 | $C_2$ | $(0,y,1/2)$ | $A_z B_{xy}$ | $\varphi_2[T_1 T_2, T_3 C_2]$ |

**No. 6:** $Pm$

This group is generated by three translations $T_{1,2,3}$ as given in Eqs. (88), and a mirror $M$:

$$M\colon (x,y,z) \to (x, -y, z). \tag{152a}$$

The $\mathbb{Z}_2$ cohomology ring is given by

$$\mathbb{Z}_2[A_m, A_x, A_y, A_z]/\langle \mathcal{R}_2 \rangle \tag{153}$$

where the relations are

$$\mathcal{R}_2\colon \quad A_x^2, \quad A_y(A_m + A_y), \quad A_z^2. \tag{154a}$$

We have the following table regarding IWPs and group cohomology at degree 3.

| Wyckoff position | Little group Intl. | Schönflies | Coordinates | LSM anomaly class | Topo. inv. |
|---|---|---|---|---|---|
| 1a | $m$ | $C_s$ | $(x, 0, z)$ | $A_x(A_m + A_y)A_z$ | $\varphi_3[T_1, T_3, M]$ |
| 1b | $m$ | $C_s$ | $(x, 1/2, z)$ | $A_x A_y A_z$ | $\varphi_3[T_1, T_3, T_2 M]$ |

**No. 7: $Pc$**

This group is generated by three translations $T_{1,2,3}$ as given in Eqs. (88), and a glide $G$:

$$G: (x, y, z) \to (x, -y, z + 1/2). \tag{155a}$$

The $\mathbb{Z}_2$ cohomology ring is given by

$$\mathbb{Z}_2[A_m, A_x, A_y]/\langle \mathscr{R}_2 \rangle \tag{156}$$

where the relations are

$$\mathscr{R}_2: \quad A_m^2, \quad A_x^2, \quad A_y(A_m + A_y). \tag{157a}$$

We have the following table regarding IWPs and group cohomology at degree 3.

| Wyckoff position | Little group Intl. | Schönflies | Coordinates | LSM anomaly class | Topo. inv. |
|---|---|---|---|---|---|
| 2a | 1 | $C_1$ | $(x, y, z), (x, -y, z + 1/2)$ | $A_m A_x A_y$ | $\widetilde{\varphi_3}[T_2, G, T_1]$ |

The expression of $\widetilde{\varphi_3}$ is given in Eq. (87).

**No. 8: $Cm$**

This group is generated by three translations $T_{1,2,3}$ as given in Eqs. (89), and a mirror $M$:

$$M: (x, y, z) \to (x, -y, z). \tag{158a}$$

The $\mathbb{Z}_2$ cohomology ring is given by

$$\mathbb{Z}_2[A_m, A_{x+y}, A_z, B_{xy}]/\langle \mathscr{R}_2, \mathscr{R}_3, \mathscr{R}_4 \rangle \tag{159}$$

where the relations are

$$\mathscr{R}_2: \quad A_m A_{x+y}, \quad A_{x+y}^2, \quad A_z^2, \tag{160a}$$
$$\mathscr{R}_3: \quad A_{x+y} B_{xy}, \tag{160b}$$
$$\mathscr{R}_4: \quad B_{xy}^2. \tag{160c}$$

We have the following table regarding IWPs and group cohomology at degree 3.

| Wyckoff position | Little group Intl. | Schönflies | Coordinates $(0,0,0) + (1/2,1/2,0)+$ | LSM anomaly class | Topo. inv. |
|---|---|---|---|---|---|
| 2a | $m$ | $C_s$ | $(x, 0, z)$ | $A_z B_{xy}$ | $\varphi_3[T_1 T_2^{-1}, T_3, M]$ |

**No. 9: $Cc$**

This group is generated by three translations $T_{1,2,3}$ as given in Eqs. (89), and a glide $G$:

$$G: (x, y, z) \to (x, -y, z + 1/2). \tag{161a}$$

The $\mathbb{Z}_2$ cohomology ring is given by

$$\mathbb{Z}_2[A_m, A_{x+y}, B_{xy}, B_{z(x+y)}]/\langle \mathcal{R}_2, \mathcal{R}_3, \mathcal{R}_4 \rangle \tag{162}$$

where the relations are

$$\mathcal{R}_2: \quad A_m A_{x+y}, \quad A_m^2, \quad A_{x+y}^2, \tag{163a}$$

$$\mathcal{R}_3: \quad A_{x+y} B_{xy}, \quad A_m B_{z(x+y)}, \quad A_m B_{xy} + A_{x+y} B_{z(x+y)}, \tag{163b}$$

$$\mathcal{R}_4: \quad B_{xy}^2, \quad B_{xy} B_{z(x+y)}, \quad B_{z(x+y)}^2. \tag{163c}$$

We have the following table regarding IWPs and group cohomology at degree 3.

| Wyckoff position | Little group Intl. | Schönflies | Coordinates $(0,0,0) + (1/2,1/2,0)+$ | LSM anomaly class | Topo. inv. |
|---|---|---|---|---|---|
| 4a | 1 | $C_1$ | $(x,y,z), \ (x,-y,z+1/2)$ | $A_m B_{xy}$ | $\widehat{\varphi_3}[T_1, T_2, G]$ |

Here the topological invariant can be chosen to be

$$\widehat{\varphi_3}[T_1, T_2, G] = \lambda(T_1, T_2, T_1^{-1} T_2^{-1} G) + \lambda(T_2, T_1, T_1^{-1} T_2^{-1} G) + \lambda(T_1, T_1^{-1} G, T_1) + \lambda(T_2, T_2^{-1} G, T_2)$$
$$+ \lambda(G, T_1, T_2) + \lambda(G, T_2, T_1). \tag{164}$$

## No. 10: $P2/m$

This group is generated by three translations $T_{1,2,3}$ as given in Eqs. (88), a two-fold rotation $C_2$, and an inversion $I$:

$$C_2: (x,y,z) \to (-x,y,-z), \tag{165a}$$
$$I: (x,y,z) \to (-x,-y,-z). \tag{165b}$$

The $\mathbb{Z}_2$ cohomology ring is given by

$$\mathbb{Z}_2[A_c, A_i, A_x, A_y, A_z]/\langle \mathcal{R}_2 \rangle \tag{166}$$

where the relations are

$$\mathcal{R}_2: \quad A_x(A_c + A_i + A_x), \quad A_y(A_i + A_y), \quad A_z(A_c + A_i + A_z). \tag{167a}$$

We have the following table regarding IWPs and group cohomology at degree 3.

| Wyckoff position | Little group Intl. | Schönflies | Coordinates | LSM anomaly class | Topo. inv. |
|---|---|---|---|---|---|
| 1a | $2/m$ | $C_{2h}$ | $(0,0,0)$ | $(A_c + A_i + A_x)(A_i + A_y)(A_c + A_i + A_z)$ | $\varphi_1[I]$ |
| 1b | $2/m$ | $C_{2h}$ | $(0,1/2,0)$ | $(A_c + A_i + A_x)A_y(A_c + A_i + A_z)$ | $\varphi_1[T_2 I]$ |
| 1c | $2/m$ | $C_{2h}$ | $(0,0,1/2)$ | $(A_c + A_i + A_x)(A_i + A_y)A_z$ | $\varphi_1[T_3 I]$ |
| 1d | $2/m$ | $C_{2h}$ | $(1/2,0,0)$ | $A_x(A_i + A_y)(A_c + A_i + A_z)$ | $\varphi_1[T_1 I]$ |
| 1e | $2/m$ | $C_{2h}$ | $(1/2,1/2,0)$ | $A_x A_y(A_c + A_i + A_z)$ | $\varphi_1[T_1 T_2 I]$ |
| 1f | $2/m$ | $C_{2h}$ | $(0,1/2,1/2)$ | $(A_c + A_i + A_x)A_y A_z$ | $\varphi_1[T_2 T_3 I]$ |
| 1g | $2/m$ | $C_{2h}$ | $(1/2,0,1/2)$ | $A_x(A_i + A_y)A_z$ | $\varphi_1[T_1 T_3 I]$ |
| 1h | $2/m$ | $C_{2h}$ | $(1/2,1/2,1/2)$ | $A_x A_y A_z$ | $\varphi_1[T_1 T_2 T_3 I]$ |

## No. 11: $P2_1/m$

This group is generated by three translations $T_{1,2,3}$ as given in Eqs. (88), a two-fold screw $S_2$, and an inversion $I$:

$$S_2: (x,y,z) \to (-x,y+1/2,-z), \tag{168a}$$
$$I: (x,y,z) \to (-x,-y,-z). \tag{168b}$$

The $\mathbb{Z}_2$ cohomology ring is given by

$$\mathbb{Z}_2[A_c, A_i, A_x, A_z]/\langle \mathcal{R}_2 \rangle \tag{169}$$

where the relations are

$$\mathcal{R}_2: \quad A_c(A_c + A_i), \quad A_x(A_c + A_i + A_x), \quad A_z(A_c + A_i + A_z). \tag{170a}$$

We have the following table regarding IWPs and group cohomology at degree 3.

| Wyckoff position | Little group Intl. | Little group Schönflies | Coordinates | LSM anomaly class | Topo. inv. |
|---|---|---|---|---|---|
| 2a | $\bar{1}$ | $C_i$ | $(0,0,0), (0,1/2,0)$ | $(A_c + A_i)(A_i + A_x)(A_i + A_z)$ | $\varphi_1[I]$ |
| 2b | $\bar{1}$ | $C_i$ | $(1/2,0,0), (1/2,1/2,0)$ | $(A_c + A_i)A_x(A_i + A_z)$ | $\varphi_1[T_1 I]$ |
| 2c | $\bar{1}$ | $C_i$ | $(0,0,1/2), (0,1/2,1/2)$ | $(A_c + A_i)(A_i + A_x)A_z$ | $\varphi_1[T_3 I]$ |
| 2d | $\bar{1}$ | $C_i$ | $(1/2,0,1/2), (1/2,1/2,1/2)$ | $(A_c + A_i)A_x A_z$ | $\varphi_1[T_1 T_3 I]$ |
| 2e | $m$ | $C_s$ | $(x,1/4,z), (-x,3/4,-z)$ | $A_c A_x A_z$ | $\varphi_3[T_1, T_3, S_2 I]$ |

### No. 12: $C2/m$

This group is generated by three translations $T_{1,2,3}$ as given in Eqs. (89), a two-fold rotation $C_2$, and an inversion $I$:

$$C_2: (x,y,z) \rightarrow (-x,y,-z), \tag{171a}$$
$$I: (x,y,z) \rightarrow (-x,-y,-z). \tag{171b}$$

The $\mathbb{Z}_2$ cohomology ring is given by

$$\mathbb{Z}_2[A_c, A_i, A_{x+y}, A_z, B_{xy}]/\langle \mathcal{R}_2, \mathcal{R}_3, \mathcal{R}_4 \rangle \tag{172}$$

where the relations are

$$\mathcal{R}_2: \quad A_c A_{x+y}, \quad A_{x+y}(A_i + A_{x+y}), \quad A_z(A_c + A_i + A_z), \tag{173a}$$
$$\mathcal{R}_3: \quad A_{x+y} B_{xy}, \tag{173b}$$
$$\mathcal{R}_4: \quad B_{xy}(A_c A_i + A_i^2 + B_{xy}). \tag{173c}$$

We have the following table regarding IWPs and group cohomology at degree 3.

| Wyckoff position | Little group Intl. | Little group Schönflies | Coordinates $(0,0,0) + (1/2,1/2,0)+$ | LSM anomaly class | Topo. inv. |
|---|---|---|---|---|---|
| 2a | $2/m$ | $C_{2h}$ | $(0,0,0)$ | $(A_c + A_i + A_z)(A_c A_i + A_i^2 + A_i A_{x+y} + B_{xy})$ | $\varphi_1[I]$ |
| 2b | $2/m$ | $C_{2h}$ | $(0,1/2,0)$ | $(A_c + A_i + A_z)B_{xy}$ | $\varphi_1[T_1 T_2 I]$ |
| 2c | $2/m$ | $C_{2h}$ | $(0,0,1/2)$ | $A_z(A_c A_i + A_i^2 + A_i A_{x+y} + B_{xy})$ | $\varphi_1[T_3 I]$ |
| 2d | $2/m$ | $C_{2h}$ | $(0,1/2,1/2)$ | $A_z B_{xy}$ | $\varphi_1[T_1 T_2 T_3 I]$ |
| 4e | $\bar{1}$ | $C_i$ | $(1/4,1/4,0), (3/4,1/4,0)$ | $A_i A_{x+y}(A_i + A_z)$ | $\varphi_1[T_1 I]$ |
| 4f | $\bar{1}$ | $C_i$ | $(1/4,1/4,1/2), (3/4,1/4,1/2)$ | $A_i A_{x+y} A_z$ | $\varphi_1[T_1 T_3 I]$ |

### No. 13: $P2/c$

This group is generated by three translations $T_{1,2,3}$ as given in Eqs. (88), a two-fold rotation $C_2$, and an inversion $I$:

$$C_2: (x,y,z) \rightarrow (-x,y,-z+1/2), \tag{174a}$$
$$I: (x,y,z) \rightarrow (-x,-y,-z). \tag{174b}$$

The $\mathbb{Z}_2$ cohomology ring is given by

$$\mathbb{Z}_2[A_c, A_i, A_x, A_y]/\langle \mathcal{R}_2 \rangle \tag{175}$$

where the relations are

$$\mathcal{R}_2: \quad A_c A_i, \quad A_x(A_c + A_i + A_x), \quad A_y(A_i + A_y). \tag{176a}$$

We have the following table regarding IWPs and group cohomology at degree 3.

| Wyckoff position | Little group Intl. | Little group Schönflies | Coordinates | LSM anomaly class | Topo. inv. |
|---|---|---|---|---|---|
| 2a | $\bar{1}$ | $C_i$ | $(0,0,0),\ (0,0,1/2)$ | $A_i(A_i+A_x)(A_i+A_y)$ | $\varphi_1[I]$ |
| 2b | $\bar{1}$ | $C_i$ | $(1/2,1/2,0),\ (1/2,1/2,1/2)$ | $A_iA_xA_y$ | $\varphi_1[T_1T_2I]$ |
| 2c | $\bar{1}$ | $C_i$ | $(0,1/2,0),\ (0,1/2,1/2)$ | $A_i(A_i+A_x)A_y$ | $\varphi_1[T_2I]$ |
| 2d | $\bar{1}$ | $C_i$ | $(1/2,0,0),\ (1/2,0,1/2)$ | $A_iA_x(A_i+A_y)$ | $\varphi_1[T_1I]$ |
| 2e | $2$ | $C_2$ | $(0,y,1/4),\ (0,-y,3/4)$ | $A_c(A_c+A_x)A_y$ | $\varphi_2[T_2,C_2]$ |
| 2f | $2$ | $C_2$ | $(1/2,y,1/4),\ (1/2,-y,3/4)$ | $A_cA_xA_y$ | $\varphi_2[T_2,T_1C_2]$ |

**No. 14:** $P2_1/c$

This group is generated by three translations $T_{1,2,3}$ as given in Eqs. (88), a two-fold screw $S_2$, and an inversion $I$:

$$S_2\colon (x,y,z) \to (-x, y+1/2, -z+1/2), \tag{177a}$$
$$I\colon (x,y,z) \to (-x,-y,-z). \tag{177b}$$

The $\mathbb{Z}_2$ cohomology ring is given by

$$\mathbb{Z}_2[A_c, A_i, A_x, B_\beta]/\langle \mathcal{R}_2, \mathcal{R}_3, \mathcal{R}_4\rangle \tag{178}$$

where the relations are

$$\mathcal{R}_2\colon\quad A_cA_i,\quad A_c^2,\quad A_x(A_c+A_i+A_x), \tag{179a}$$
$$\mathcal{R}_3\colon\quad A_cB_\beta, \tag{179b}$$
$$\mathcal{R}_4\colon\quad B_\beta(A_i^2+B_\beta). \tag{179c}$$

We have the following table regarding IWPs and group cohomology at degree 3.

| Wyckoff position | Little group Intl. | Little group Schönflies | Coordinates | LSM anomaly class | Topo. inv. |
|---|---|---|---|---|---|
| 2a | $\bar{1}$ | $C_i$ | $(0,0,0),\ (0,1/2,1/2)$ | $(A_i+A_x)B_\beta$ | $\varphi_1[I]$ |
| 2b | $\bar{1}$ | $C_i$ | $(1/2,0,0),\ (1/2,1/2,1/2)$ | $A_xB_\beta$ | $\varphi_1[T_1I]$ |
| 2c | $\bar{1}$ | $C_i$ | $(0,0,1/2),\ (0,1/2,0)$ | $(A_i+A_x)(A_i^2+B_\beta)$ | $\varphi_1[T_2I]$ |
| 2d | $\bar{1}$ | $C_i$ | $(1/2,0,1/2),\ (1/2,1/2,0)$ | $A_x(A_i^2+B_\beta)$ | $\varphi_1[T_1T_2I]$ |

**No. 15:** $C2/c$

This group is generated by three translations $T_{1,2,3}$ as given in Eqs. (89), a two-fold rotation $C_2$, and an inversion $I$:

$$C_2\colon (x,y,z) \to (-x, y, -z+1/2), \tag{180a}$$
$$I\colon (x,y,z) \to (-x,-y,-z). \tag{180b}$$

The $\mathbb{Z}_2$ cohomology ring is given by

$$\mathbb{Z}_2[A_c, A_i, A_{x+y}, B_{xy}, B_{z(x+y)}]/\langle \mathcal{R}_2, \mathcal{R}_3, \mathcal{R}_4\rangle \tag{181}$$

where the relations are

$$\mathcal{R}_2\colon\quad A_cA_i,\quad A_cA_{x+y},\quad A_{x+y}(A_i+A_{x+y}), \tag{182a}$$
$$\mathcal{R}_3\colon\quad A_{x+y}B_{xy},\quad A_cB_{z(x+y)},\quad A_iB_{xy}+A_iB_{z(x+y)}+A_{x+y}B_{z(x+y)}, \tag{182b}$$
$$\mathcal{R}_4\colon\quad B_{xy}(A_i^2+B_{xy}),\quad B_{xy}(A_i^2+B_{z(x+y)}),\quad B_{z(x+y)}(A_i^2+B_{z(x+y)}). \tag{182c}$$

We have the following table regarding IWPs and group cohomology at degree 3.

| Wyckoff position | Little group Intl. | Little group Schönflies | Coordinates $(0,0,0)+(1/2,1/2,0)+$ | LSM anomaly class | Topo. inv. |
|---|---|---|---|---|---|
| 4a | $\bar{1}$ | $C_i$ | $(0,0,0),\ (0,0,1/2)$ | $A_i(A_i^2+A_iA_{x+y}+B_{xy})$ | $\varphi_1[I]$ |
| 4b | $\bar{1}$ | $C_i$ | $(0,1/2,0),\ (0,1/2,1/2)$ | $A_iB_{xy}$ | $\varphi_1[T_1T_2I]$ |
| 4c | $\bar{1}$ | $C_i$ | $(1/4,1/4,0),\ (3/4,1/4,1/2)$ | $A_i(B_{xy}+B_{z(x+y)})$ | $\varphi_1[T_1I]$ |
| 4d | $\bar{1}$ | $C_i$ | $(1/4,1/4,1/2),\ (3/4,1/4,0)$ | $A_i(A_iA_{x+y}+B_{xy}+B_{z(x+y)})$ | $\varphi_1[T_1T_3I]$ |
| 4e | $2$ | $C_2$ | $(0,y,1/4),\ (0,-y,3/4)$ | $A_cB_{xy}$ | $\varphi_2[T_1T_2,C_2]$ |

**No. 16:** $P222$

This group is generated by three translations $T_{1,2,3}$ as given in Eqs. (88), a two-fold rotation $C_2$, and a two-fold rotation $C_2'$:

$$C_2\colon (x,y,z) \to (-x,-y,z), \tag{183a}$$
$$C_2'\colon (x,y,z) \to (-x,y,-z). \tag{183b}$$

The $\mathbb{Z}_2$ cohomology ring is given by

$$\mathbb{Z}_2[A_c, A_{c'}, A_x, A_y, A_z]/\langle \mathcal{R}_2 \rangle \tag{184}$$

where the relations are

$$\mathcal{R}_2\colon \quad A_x(A_c + A_{c'} + A_x), \quad A_y(A_c + A_y), \quad A_z(A_{c'} + A_z). \tag{185a}$$

We have the following table regarding IWPs and group cohomology at degree 3.

| Wyckoff position | Little group Intl. | Little group Schönflies | Coordinates | LSM anomaly class | Topo. inv. |
|---|---|---|---|---|---|
| 1a | 222 | $D_2$ | $(0,0,0)$ | $(A_c + A_{c'} + A_x)(A_c + A_y)(A_{c'} + A_z)$ | $\varphi_2[C_2, C_2']$ |
| 1b | 222 | $D_2$ | $(1/2,0,0)$ | $A_x(A_c + A_y)(A_{c'} + A_z)$ | $\varphi_2[T_1C_2, T_1C_2']$ |
| 1c | 222 | $D_2$ | $(0,1/2,0)$ | $(A_c + A_{c'} + A_x)A_y(A_{c'} + A_z)$ | $\varphi_2[T_2C_2, C_2']$ |
| 1d | 222 | $D_2$ | $(0,0,1/2)$ | $(A_c + A_{c'} + A_x)(A_c + A_y)A_z$ | $\varphi_2[C_2, T_3C_2']$ |
| 1e | 222 | $D_2$ | $(1/2,1/2,0)$ | $A_xA_y(A_{c'} + A_z)$ | $\varphi_2[T_1T_2C_2, T_1C_2']$ |
| 1f | 222 | $D_2$ | $(1/2,0,1/2)$ | $A_x(A_c + A_y)A_z$ | $\varphi_2[T_1C_2, T_1T_3C_2']$ |
| 1g | 222 | $D_2$ | $(0,1/2,1/2)$ | $(A_c + A_{c'} + A_x)A_yA_z$ | $\varphi_2[T_2C_2, T_3C_2']$ |
| 1h | 222 | $D_2$ | $(1/2,1/2,1/2)$ | $A_xA_yA_z$ | $\varphi_2[T_1T_2C_2, T_1T_3C_2']$ |

**No. 17:** $P222_1$

This group is generated by three translations $T_{1,2,3}$ as given in Eqs. (88), a two-fold screw $S_2$, and a two-fold rotation $C_2'$:

$$S_2\colon (x,y,z) \to (-x,-y,z+1/2), \tag{186a}$$
$$C_2'\colon (x,y,z) \to (-x,y,-z+1/2). \tag{186b}$$

The $\mathbb{Z}_2$ cohomology ring is given by

$$\mathbb{Z}_2[A_c, A_{c'}, A_x, A_y]/\langle \mathcal{R}_2 \rangle \tag{187}$$

where the relations are

$$\mathcal{R}_2\colon \quad A_c(A_c + A_{c'}), \quad A_x(A_c + A_{c'} + A_x), \quad A_y(A_c + A_y). \tag{188a}$$

We have the following table regarding IWPs and group cohomology at degree 3.

| Wyckoff position | Little group Intl. | Little group Schönflies | Coordinates | LSM anomaly class | Topo. inv. |
|---|---|---|---|---|---|
| 2a | 2 | $C_2$ | $(x,0,0), (-x,0,1/2)$ | $A_cA_x(A_{c'} + A_y)$ | $\varphi_2[T_1, S_2C_2']$ |
| 2b | 2 | $C_2$ | $(x,1/2,0), (-x,1/2,1/2)$ | $A_cA_xA_y$ | $\varphi_2[T_1, T_2S_2C_2']$ |
| 2c | 2 | $C_2$ | $(0,y,1/4), (0,-y,3/4)$ | $(A_c + A_{c'})(A_{c'} + A_x)A_y$ | $\varphi_2[T_2, C_2']$ |
| 2d | 2 | $C_2$ | $(1/2,y,1/4), (1/2,-y,3/4)$ | $(A_c + A_{c'})A_xA_y$ | $\varphi_2[T_2, T_1C_2']$ |

**No. 18:** $P2_12_12$

This group is generated by three translations $T_{1,2,3}$ as given in Eqs. (88), a two-fold rotation $C_2$, and a two-fold screw $S_2'$:

$$C_2\colon (x,y,z) \to (-x,-y,z), \tag{189a}$$
$$S_2'\colon (x,y,z) \to (-x+1/2,y+1/2,-z). \tag{189b}$$

The $\mathbb{Z}_2$ cohomology ring is given by

$$\mathbb{Z}_2[A_c, A_{c'}, A_z, B_\beta]/\langle \mathcal{R}_2, \mathcal{R}_3, \mathcal{R}_4\rangle \tag{190}$$

where the relations are

$$\mathcal{R}_2: \quad A_c A_{c'}, \quad A_{c'}^2, \quad A_z(A_{c'} + A_z), \tag{191a}$$
$$\mathcal{R}_3: \quad A_{c'} B_\beta, \tag{191b}$$
$$\mathcal{R}_4: \quad B_\beta(A_c^2 + B_\beta). \tag{191c}$$

We have the following table regarding IWPs and group cohomology at degree 3.

| Wyckoff position | Little group Intl. | Little group Schönflies | Coordinates | LSM anomaly class | Topo. inv. |
|---|---|---|---|---|---|
| 2a | 2 | $C_2$ | $(0,0,z)$, $(1/2,1/2,-z)$ | $A_z(A_c^2 + B_\beta)$ | $\varphi_2[T_3, C_2]$ |
| 2b | 2 | $C_2$ | $(0,1/2,z)$, $(1/2,0,-z)$ | $A_z B_\beta$ | $\varphi_2[T_3, T_2 C_2]$ |

## No. 19: $P2_12_12_1$

This group is generated by three translations $T_{1,2,3}$ as given in Eqs. (88), a two-fold screw $S_2$, and a two-fold screw $S_2'$:

$$S_2: (x,y,z) \to (-x+1/2, -y, z+1/2), \tag{192a}$$
$$S_2': (x,y,z) \to (-x, y+1/2, -z+1/2). \tag{192b}$$

The $\mathbb{Z}_2$ cohomology ring is given by

$$\mathbb{Z}_2[A_c, A_{c'}, B_{\beta 1}, B_{\beta 2}]/\langle \mathcal{R}_2, \mathcal{R}_3, \mathcal{R}_4\rangle \tag{193}$$

where the relations are

$$\mathcal{R}_2: \quad A_c A_{c'}, \quad A_c^2, \quad A_{c'}^2, \tag{194a}$$
$$\mathcal{R}_3: \quad (A_c + A_{c'})B_{\beta 1}, \quad A_c(B_{\beta 1} + B_{\beta 2}), \quad A_{c'} B_{\beta 2}, \tag{194b}$$
$$\mathcal{R}_4: \quad B_{\beta 1}^2, \quad B_{\beta 1} B_{\beta 2}, \quad B_{\beta 2}^2. \tag{194c}$$

We have the following table regarding IWPs and group cohomology at degree 3.

| Wyckoff position | Little group Intl. | Little group Schönflies | Coordinates | LSM anomaly class | Topo. inv. |
|---|---|---|---|---|---|
| 4a | 1 | $C_1$ | $(x,y,z)$, $(-x+1/2, -y, z+1/2)$, $(-x, y+1/2, -z+1/2)$, $(x+1/2, -y+1/2, -z)$ | $A_c B_{\beta 1}$ | $\widehat{\varphi_4}[T_2, T_3, S_2, S_2']$ |

Here the topological invariant can be chosen to be

$$
\begin{aligned}
\widehat{\varphi_4}[T_2, T_3, S_2, S_2'] =& \lambda(T_3, T_2, T_2^{-1}) + \lambda(T_2, T_3, T_2^{-1}) + \lambda(T_2, T_2^{-1}, T_3) + \lambda(T_2, T_3, T_3^{-1}) + \lambda(T_3, T_2, T_3^{-1}) + \lambda(T_3, T_3^{-1}, T_2) \\
&+ \lambda(T_2, T_2^{-1}T_3^{-1}S_2', T_2^{-1}T_3^{-1}S_2') + \lambda(T_3^{-1}S_2', T_2, T_2^{-1}T_3^{-1}S_2') + \lambda(T_3^{-1}S_2', T_3^{-1}S_2', T_2) \\
&+ \lambda(T_3, T_2^{-1}T_3^{-1}S_2, T_2^{-1}T_3^{-1}S_2) + \lambda(T_2^{-1}S_2, T_3, T_2^{-1}T_3^{-1}S_2) + \lambda(T_2^{-1}S_2, T_2^{-1}S_2, T_3) \\
&+ \lambda(T_3, 1, T_3) + \lambda(T_2, 1, T_2).
\end{aligned}
\tag{195}
$$

## No. 20: $C222_1$

This group is generated by three translations $T_{1,2,3}$ as given in Eqs. (89), a two-fold screw $S_2$, and a two-fold rotation $C_2'$:

$$S_2: (x,y,z) \to (-x, -y, z+1/2), \tag{196a}$$
$$C_2': (x,y,z) \to (-x, y, -z+1/2). \tag{196b}$$

The $\mathbb{Z}_2$ cohomology ring is given by

$$\mathbb{Z}_2[A_c, A_{c'}, A_{x+y}, B_{xy}]/\langle \mathcal{R}_2, \mathcal{R}_3, \mathcal{R}_4\rangle \tag{197}$$

where the relations are

$$\mathcal{R}_2: \quad A_{c'}A_{x+y}, \quad A_c(A_c + A_{c'}), \quad A_{x+y}(A_c + A_{x+y}), \tag{198a}$$
$$\mathcal{R}_3: \quad A_{x+y}B_{xy}, \tag{198b}$$
$$\mathcal{R}_4: \quad B_{xy}^2. \tag{198c}$$

We have the following table regarding IWPs and group cohomology at degree 3.

| Wyckoff position | Little group Intl. | Schönflies | Coordinates $(0,0,0)+(1/2,1/2,0)+$ | LSM anomaly class | Topo. inv. |
|---|---|---|---|---|---|
| 4a | 2 | $C_2$ | $(x,0,0)$, $(-x,0,1/2)$ | $A_c B_{xy}$ | $\varphi_2[T_1 T_2^{-1}, S_2 C_2']$ |
| 4b | 2 | $C_2$ | $(0,y,1/4)$, $(0,-y,3/4)$ | $(A_c + A_{c'})B_{xy}$ | $\varphi_2[T_1 T_2, C_2']$ |

**No. 21:** $C222$

This group is generated by three translations $T_{1,2,3}$ as given in Eqs. (89), a two-fold rotation $C_2$, and a two-fold rotation $C_2'$:

$$C_2: (x,y,z) \to (-x,-y,z), \tag{199a}$$
$$C_2': (x,y,z) \to (-x,y,-z). \tag{199b}$$

The $\mathbb{Z}_2$ cohomology ring is given by

$$\mathbb{Z}_2[A_c, A_{c'}, A_{x+y}, A_z, B_{xy}]/\langle \mathcal{R}_2, \mathcal{R}_3, \mathcal{R}_4\rangle \tag{200}$$

where the relations are

$$\mathcal{R}_2: \quad A_{c'}A_{x+y}, \quad A_{x+y}(A_c + A_{x+y}), \quad A_z(A_{c'} + A_z), \tag{201a}$$
$$\mathcal{R}_3: \quad A_{x+y}B_{xy}, \tag{201b}$$
$$\mathcal{R}_4: \quad B_{xy}(A_c^2 + A_c A_{c'} + B_{xy}). \tag{201c}$$

We have the following table regarding IWPs and group cohomology at degree 3.

| Wyckoff position | Little group Intl. | Schönflies | Coordinates $(0,0,0)+(1/2,1/2,0)+$ | LSM anomaly class | Topo. inv. |
|---|---|---|---|---|---|
| 2a | 222 | $D_2$ | $(0,0,0)$ | $(A_{c'} + A_z)(A_c^2 + A_c A_{c'} + A_c A_{x+y} + B_{xy})$ | $\varphi_2[C_2, C_2']$ |
| 2b | 222 | $D_2$ | $(0,1/2,0)$ | $(A_{c'} + A_z)B_{xy}$ | $\varphi_2[T_1 T_2 C_2, C_2']$ |
| 2c | 222 | $D_2$ | $(1/2,0,1/2)$ | $A_z B_{xy}$ | $\varphi_2[T_1 T_2 C_2, T_3 C_2']$ |
| 2d | 222 | $D_2$ | $(0,0,1/2)$ | $A_z(A_c^2 + A_c A_{c'} + A_c A_{x+y} + B_{xy})$ | $\varphi_2[C_2, T_3 C_2']$ |
| 4k | 2 | $C_2$ | $(1/4,1/4,z)$, $(3/4,1/4,-z)$ | $A_c A_{x+y} A_z$ | $\varphi_2[T_3, T_1 C_2]$ |

**No. 22:** $F222$

This group is generated by three translations $T_{1,2,3}$ as given in Eqs. (92), a two-fold rotation $C_2$, and a two-fold rotation $C_2'$:

$$C_2: (x,y,z) \to (-x,-y,z), \tag{202a}$$
$$C_2': (x,y,z) \to (-x,y,-z). \tag{202b}$$

The $\mathbb{Z}_2$ cohomology ring is given by

$$\mathbb{Z}_2[A_c, A_{c'}, A_{x+y}, A_{x+z}, C_\gamma, C_{xyz}]/\langle \mathcal{R}_2, \mathcal{R}_4, \mathcal{R}_6\rangle \tag{203}$$

where the relations are

$$\mathscr{R}_2\colon \quad A_c A_{x+y} + A_{c'} A_{x+z}, \quad A_{x+y}(A_{c'} + A_{x+y}), \quad A_{x+z}(A_c + A_{x+z}), \tag{204a}$$

$$\mathscr{R}_4\colon \quad A_c C_\gamma + A_{x+z} C_\gamma + A_c C_{xyz}, \quad A_{c'} C_\gamma + A_{x+y} C_\gamma + A_{c'} C_{xyz}, \quad A_{x+y} C_{xyz}, \quad A_{x+z} C_{xyz}, \tag{204b}$$

$$\mathscr{R}_6\colon \quad C_\gamma(A_c^2 A_{c'} + A_c A_{c'}^2 + C_\gamma), \quad C_{xyz}(A_c^2 A_{c'} + A_c A_{c'}^2 + C_\gamma), \quad C_{xyz}(A_c^2 A_{c'} + A_c A_{c'}^2 + C_{xyz}). \tag{204c}$$

We have the following table regarding IWPs and group cohomology at degree 3.

| Wyckoff position | Little group | | Coordinates $(0,0,0)+$ $(0,1/2,1/2)+$ $(1/2,0,1/2)+$ $(1/2,1/2,0)+$ | LSM anomaly class | Topo. inv. |
|---|---|---|---|---|---|
| | Intl. | Schönflies | | | |
| 4a | 222 | $D_2$ | $(0,0,0)$ | $A_c(A_c + A_{c'})(A_{c'} + A_{x+y}) + C_{xyz}$ | $\varphi_2[C_2, C_2']$ |
| 4b | 222 | $D_2$ | $(0,0,1/2)$ | $C_{xyz}$ | $\varphi_2[C_2, T_1 T_2 T_3^{-1} C_2']$ |
| 4c | 222 | $D_2$ | $(1/4,1/4,1/4)$ | $C_\gamma + C_{xyz}$ | $\varphi_2[T_3 C_2, T_2 C_2']$ |
| 4d | 222 | $D_2$ | $(1/4,1/4,3/4)$ | $A_c A_{c'}(A_{x+y} + A_{x+z}) + C_\gamma + C_{xyz}$ | $\varphi_2[T_1^{-1} T_2 C_2, T_2 C_2']$ |

### No. 23: $I222$

This group is generated by three translations $T_{1,2,3}$ as given in Eqs. (91), a two-fold rotation $C_2$, and a two-fold rotation $C_2'$:

$$C_2\colon (x,y,z) \to (-x,-y,z), \tag{205a}$$

$$C_2'\colon (x,y,z) \to (-x,y,-z). \tag{205b}$$

The $\mathbb{Z}_2$ cohomology ring is given by

$$\mathbb{Z}_2[A_c, A_{c'}, A_{x+y+z}, B_\beta, B_{x(y+z)}, B_{y(x+z)}, C_{xyz}]/\langle \mathscr{R}_2, \mathscr{R}_3, \mathscr{R}_4, \mathscr{R}_5, \mathscr{R}_6 \rangle \tag{206}$$

where the relations are

$$\mathscr{R}_2\colon \quad A_c A_{x+y+z}, \quad A_{c'} A_{x+y+z}, \quad A_{x+y+z}^2, \tag{207a}$$

$$\mathscr{R}_3\colon \quad A_{x+y+z} B_\beta, \quad A_{x+y+z} B_{x(y+z)}, \quad A_c B_\beta + A_c B_{x(y+z)} + A_c B_{y(x+z)} + A_{c'} B_{y(x+z)}, \quad A_{x+y+z} B_{y(x+z)}, \tag{207b}$$

$$\mathscr{R}_4\colon \quad A_{x+y+z} C_{xyz}, \quad A_c^2 B_\beta + A_{c'}^2 B_\beta + B_\beta^2 + A_c A_{c'} B_{x(y+z)}, \quad A_c A_{c'} B_\beta + B_\beta B_{x(y+z)} + A_c C_{xyz} + A_{c'} C_{xyz},$$

$$\quad A_c A_{c'} B_\beta + A_c^2 B_{x(y+z)} + A_c^2 B_{y(x+z)} + B_\beta B_{y(x+z)} + A_c C_{xyz}, \quad B_{x(y+z)}(A_c A_{c'} + B_{x(y+z)}),$$

$$\quad A_c^2 B_\beta + A_c^2 B_{x(y+z)} + A_c^2 B_{y(x+z)} + B_{x(y+z)} B_{y(x+z)} + A_c C_{xyz}, \quad A_c^2 B_\beta + B_{y(x+z)}^2, \tag{207c}$$

$$\mathscr{R}_5\colon \quad (A_c^2 + A_{c'}^2 + B_\beta) C_{xyz}, \quad B_{x(y+z)} C_{xyz}, \quad (A_c^2 + A_c A_{c'} + B_{y(x+z)}) C_{xyz}, \tag{207d}$$

$$\mathscr{R}_6\colon \quad C_{xyz}(A_c^2 A_{c'} + A_c A_{c'}^2 + C_{xyz}). \tag{207e}$$

We have the following table regarding IWPs and group cohomology at degree 3.

| Wyckoff position | Little group | | Coordinates $(0,0,0)+$ $(1/2,1/2,1/2)+$ | LSM anomaly class | Topo. inv. |
|---|---|---|---|---|---|
| | Intl. | Schönflies | | | |
| 2a | 222 | $D_2$ | $(0,0,0)$ | $(A_c + A_{c'})(A_c A_{c'} + B_{x(y+z)}) + C_{xyz}$ | $\varphi_2[C_2, C_2']$ |
| 2b | 222 | $D_2$ | $(1/2,0,0)$ | $C_{xyz}$ | $\varphi_2[T_2 T_3 C_2, T_2 T_3 C_2']$ |
| 2c | 222 | $D_2$ | $(0,0,1/2)$ | $A_c B_\beta + A_c B_{y(x+z)} + C_{xyz}$ | $\varphi_2[C_2, T_1 T_2 C_2']$ |
| 2d | 222 | $D_2$ | $(0,1/2,0)$ | $A_{c'}(B_{x(y+z)} + B_{y(x+z)}) + C_{xyz}$ | $\varphi_2[T_1 T_3 C_2, C_2']$ |

### No. 24: $I2_1 2_1 2_1$

This group is generated by three translations $T_{1,2,3}$ as given in Eqs. (91), a two-fold rotation $C_2$, and a two-fold rotation $C_2'$:

$$C_2\colon (x,y,z) \to (-x,-y+1/2,z), \tag{208a}$$

$$C_2'\colon (x,y,z) \to (-x+1/2,y,-z). \tag{208b}$$

The $\mathbb{Z}_2$ cohomology ring is given by

$$\mathbb{Z}_2[A_c, A_{c'}, A_{x+y+z}, B_{y(x+z)}, B_{z(x+y)}]/\langle \mathcal{R}_2, \mathcal{R}_3, \mathcal{R}_4 \rangle \tag{209}$$

where the relations are

$$\mathcal{R}_2: \quad A_c(A_{c'} + A_{x+y+z}), \quad A_{c'}(A_c + A_{x+y+z}), \quad A_cA_{c'} + A_{x+y+z}^2, \tag{210a}$$

$$\mathcal{R}_3: \quad (A_{c'} + A_{x+y+z})B_{y(x+z)}, \quad A_{c'}B_{y(x+z)} + A_cB_{z(x+y)}, \quad A_{c'}B_{y(x+z)} + A_{x+y+z}B_{z(x+y)}, \tag{210b}$$

$$\mathcal{R}_4: \quad A_c^3A_{c'} + B_{y(x+z)}^2, \quad A_c^3A_{c'} + B_{y(x+z)}B_{z(x+y)}, \quad A_c^3A_{c'} + B_{z(x+y)}^2. \tag{210c}$$

We have the following table regarding IWPs and group cohomology at degree 3.

| Wyckoff position | Little group Intl. | Little group Schönflies | Coordinates $(0,0,0)+$ $(1/2,1/2,1/2)+$ | LSM anomaly class | Topo. inv. |
|---|---|---|---|---|---|
| 4a | 2 | $C_2$ | $(x,0,1/4), (-x+1/2,0,3/4)$ | $A_{c'}(A_c^2 + B_{y(x+z)})$ | $\varphi_2[T_2T_3, T_2C_2C_2']$ |
| 4b | 2 | $C_2$ | $(1/4,y,0), (1/4,-y,1/2)$ | $A_{c'}(B_{y(x+z)} + B_{z(x+y)})$ | $\varphi_2[T_1T_3, C_2']$ |
| 4c | 2 | $C_2$ | $(0,1/4,z), (0,3/4,-z+1/2)$ | $(A_c + A_{c'})B_{y(x+z)}$ | $\varphi_2[T_1T_2, C_2]$ |

### No. 25: $Pmm2$

This group is generated by three translations $T_{1,2,3}$ as given in Eqs. (88), a two-fold rotation $C_2$, and a mirror $M$:

$$C_2: (x,y,z) \to (-x,-y,z), \tag{211a}$$

$$M: (x,y,z) \to (x,-y,z). \tag{211b}$$

The $\mathbb{Z}_2$ cohomology ring is given by

$$\mathbb{Z}_2[A_c, A_m, A_x, A_y, A_z]/\langle \mathcal{R}_2 \rangle \tag{212}$$

where the relations are

$$\mathcal{R}_2: \quad A_x(A_c + A_x), \quad A_y(A_c + A_m + A_y), \quad A_z^2. \tag{213a}$$

We have the following table regarding IWPs and group cohomology at degree 3.

| Wyckoff position | Little group Intl. | Little group Schönflies | Coordinates | LSM anomaly class | Topo. inv. |
|---|---|---|---|---|---|
| 1a | $mm2$ | $C_{2v}$ | $(0,0,z)$ | $(A_c + A_x)(A_c + A_m + A_y)A_z$ | $\varphi_2[T_3, C_2]$ |
| 1b | $mm2$ | $C_{2v}$ | $(0,1/2,z)$ | $(A_c + A_x)A_yA_z$ | $\varphi_2[T_3, T_2C_2]$ |
| 1c | $mm2$ | $C_{2v}$ | $(1/2,0,z)$ | $A_x(A_c + A_m + A_y)A_z$ | $\varphi_2[T_3, T_1C_2]$ |
| 1d | $mm2$ | $C_{2v}$ | $(1/2,1/2,z)$ | $A_xA_yA_z$ | $\varphi_2[T_3, T_1T_2C_2]$ |

### No. 26: $Pmc2_1$

This group is generated by three translations $T_{1,2,3}$ as given in Eqs. (88), a two-fold screw $S_2$, and a glide $G$:

$$S_2: (x,y,z) \to (-x,-y,z+1/2), \tag{214a}$$

$$G: (x,y,z) \to (x,-y,z+1/2). \tag{214b}$$

The $\mathbb{Z}_2$ cohomology ring is given by

$$\mathbb{Z}_2[A_c, A_m, A_x, A_y]/\langle \mathcal{R}_2 \rangle \tag{215}$$

where the relations are

$$\mathcal{R}_2: \quad (A_c + A_m)^2, \quad A_x(A_c + A_x), \quad A_y(A_c + A_m + A_y). \tag{216a}$$

We have the following table regarding IWPs and group cohomology at degree 3.

| Wyckoff position | Little group Intl. | Little group Schönflies | Coordinates | LSM anomaly class | Topo. inv. |
|---|---|---|---|---|---|
| 2a | $m$ | $C_s$ | $(0, y, z)$, $(0, -y, z + 1/2)$ | $(A_c + A_m)(A_c + A_x)A_y$ | $\widetilde{\varphi_3}[T_2, G, T_3^{-1}S_2 G]$ |
| 2b | $m$ | $C_s$ | $(1/2, y, z)$, $(1/2, -y, z + 1/2)$ | $(A_c + A_m)A_x A_y$ | $\widetilde{\varphi_3}[T_2, G, T_1 T_3^{-1}S_2 G]$ |

The expression of $\widetilde{\varphi_3}$ is given in Eq. (87).

## No. 27: $Pcc2$

This group is generated by three translations $T_{1,2,3}$ as given in Eqs. (88), a two-fold rotation $C_2$, and a glide $G$:

$$C_2\colon (x, y, z) \to (-x, -y, z), \tag{217a}$$
$$G\colon (x, y, z) \to (x, -y, z + 1/2). \tag{217b}$$

The $\mathbb{Z}_2$ cohomology ring is given by

$$\mathbb{Z}_2[A_c, A_m, A_x, A_y]/\langle \mathscr{R}_2 \rangle \tag{218}$$

where the relations are

$$\mathscr{R}_2\colon \quad A_m^2, \quad A_x(A_c + A_x), \quad A_y(A_c + A_m + A_y). \tag{219a}$$

We have the following table regarding IWPs and group cohomology at degree 3.

| Wyckoff position | Little group Intl. | Little group Schönflies | Coordinates | LSM anomaly class | Topo. inv. |
|---|---|---|---|---|---|
| 2a | 2 | $C_2$ | $(0, 0, z)$, $(0, 0, z + 1/2)$ | $A_m(A_c + A_x)(A_c + A_y)$ | $\varphi_2[G, C_2]$ |
| 2b | 2 | $C_2$ | $(0, 1/2, z)$, $(0, 1/2, z + 1/2)$ | $A_m(A_c + A_x)A_y$ | $\varphi_2[C_2 G, T_2 C_2]$ |
| 2c | 2 | $C_2$ | $(1/2, 0, z)$, $(1/2, 0, z + 1/2)$ | $A_m A_x(A_c + A_y)$ | $\varphi_2[G, T_1 C_2]$ |
| 2d | 2 | $C_2$ | $(1/2, 1/2, z)$, $(1/2, 1/2, z + 1/2)$ | $A_m A_x A_y$ | $\varphi_2[T_2 G, T_1 T_2 C_2]$ |

## No. 28: $Pma2$

This group is generated by three translations $T_{1,2,3}$ as given in Eqs. (88), a two-fold rotation $C_2$, and a glide $G$:

$$C_2\colon (x, y, z) \to (-x, -y, z), \tag{220a}$$
$$G\colon (x, y, z) \to (x + 1/2, -y, z). \tag{220b}$$

The $\mathbb{Z}_2$ cohomology ring is given by

$$\mathbb{Z}_2[A_c, A_m, A_y, A_z]/\langle \mathscr{R}_2 \rangle \tag{221}$$

where the relations are

$$\mathscr{R}_2\colon \quad A_m(A_c + A_m), \quad A_y(A_c + A_m + A_y), \quad A_z^2. \tag{222a}$$

We have the following table regarding IWPs and group cohomology at degree 3.

| Wyckoff position | Little group Intl. | Little group Schönflies | Coordinates | LSM anomaly class | Topo. inv. |
|---|---|---|---|---|---|
| 2a | 2 | $C_2$ | $(0, 0, z)$, $(1/2, 0, z)$ | $(A_c + A_m)(A_c + A_y)A_z$ | $\varphi_2[T_3, C_2]$ |
| 2b | 2 | $C_2$ | $(0, 1/2, z)$, $(1/2, 1/2, z)$ | $(A_c + A_m)A_y A_z$ | $\varphi_2[T_3, T_2 C_2]$ |
| 2c | $m$ | $C_s$ | $(1/4, y, z)$, $(3/4, -y, z)$ | $A_m A_y A_z$ | $\varphi_3[T_2, T_3, C_2 G]$ |

**No. 29:** $Pca2_1$

This group is generated by three translations $T_{1,2,3}$ as given in Eqs. (88), a two-fold screw $S_2$, and a glide $G$:

$$S_2\colon (x,y,z) \to (-x,-y,z+1/2), \tag{223a}$$
$$G\colon (x,y,z) \to (x+1/2,-y,z). \tag{223b}$$

The $\mathbb{Z}_2$ cohomology ring is given by

$$\mathbb{Z}_2[A_c, A_m, A_y]/\langle \mathscr{R}_2 \rangle \tag{224}$$

where the relations are

$$\mathscr{R}_2\colon \quad A_c^2, \quad A_m(A_c + A_m), \quad A_y(A_c + A_m + A_y). \tag{225a}$$

We have the following table regarding IWPs and group cohomology at degree 3.

| Wyckoff position | Little group Intl. | Little group Schönflies | Coordinates | LSM anomaly class | Topo. inv. |
|---|---|---|---|---|---|
| 4a | 1 | $C_1$ | $(x,y,z)$, $(-x,-y,z+1/2)$, $(x+1/2,-y,z)$, $(-x+1/2,y,z+1/2)$ | $A_c A_m A_y$ | $\widehat{\varphi_4}[T_1, T_2, T_3^{-1}S_2, G]$ |

Here the topological invariant can be chosen to be

$$\begin{aligned}
\widehat{\varphi_4}[T_1,T_2,T_3^{-1}S_2,G] =\ & \lambda((T_2,T_1,T_1^{-1}) + \lambda(T_1,T_2,T_1^{-1}) + \lambda(T_1,T_1^{-1},T_2) + \lambda(T_2,1,T_2) \\
& + \lambda(T_2,T_1^{-1}T_2^{-1}G,T_1^{-1}T_2^{-1}G) + \lambda(T_1^{-1}G,T_2,T_1^{-1}T_2^{-1}G) + \lambda(T_1^{-1}G,T_1^{-1}G,T_2) \\
& + \lambda(T_1,T_2,T_3^{-1}S_2G) + \lambda(T_2,T_1,T_3^{-1}S_2G) + \lambda(T_1,T_3^{-1}S_2G,T_2) \\
& + \lambda(T_3^{-1}S_2,T_1^{-1}G,T_2) + \lambda(T_1^{-1}G,T_3^{-1}S_2,T_2) + \lambda(T_3^{-1}S_2,T_2,T_1^{-1}T_2^{-1}G)) + \lambda(T_1^{-1}G,T_2,T_2^{-1}T_3^{-1}S_2) \\
& + \lambda(T_2,T_2^{-1}T_3^{-1}S_2,T_1^{-1}T_2^{-1}G) + \lambda(T_2,T_1^{-1}T_2^{-1}G,T_2^{-1}T_3^{-1}S_2).
\end{aligned} \tag{226}$$

**No. 30:** $Pnc2$

This group is generated by three translations $T_{1,2,3}$ as given in Eqs. (88), a two-fold rotation $C_2$, and a glide $G$:

$$C_2\colon (x,y,z) \to (-x,-y,z), \tag{227a}$$
$$G\colon (x,y,z) \to (x,-y+1/2,z+1/2). \tag{227b}$$

The $\mathbb{Z}_2$ cohomology ring is given by

$$\mathbb{Z}_2[A_c, A_m, A_x, B_\beta]/\langle \mathscr{R}_2, \mathscr{R}_3, \mathscr{R}_4 \rangle \tag{228}$$

where the relations are

$$\mathscr{R}_2\colon \quad A_c A_m, \quad A_m^2, \quad A_x(A_c + A_x), \tag{229a}$$
$$\mathscr{R}_3\colon \quad A_m B_\beta, \tag{229b}$$
$$\mathscr{R}_4\colon \quad B_\beta^2. \tag{229c}$$

We have the following table regarding IWPs and group cohomology at degree 3.

| Wyckoff position | Little group Intl. | Little group Schönflies | Coordinates | LSM anomaly class | Topo. inv. |
|---|---|---|---|---|---|
| 2a | 2 | $C_2$ | $(0,0,z)$, $(0,1/2,z+1/2)$ | $(A_c + A_x)B_\beta$ | $\varphi_2[T_3, C_2]$ |
| 2b | 2 | $C_2$ | $(1/2,0,z)$, $(1/2,1/2,z+1/2)$ | $A_x B_\beta$ | $\varphi_2[T_3, T_1 C_2]$ |

**No. 31: $Pmn2_1$**

This group is generated by three translations $T_{1,2,3}$ as given in Eqs. (88), a two-fold screw $S_2$, and a glide $G$:

$$S_2\colon (x,y,z) \to (-x+1/2, -y, z+1/2), \tag{230a}$$
$$G\colon (x,y,z) \to (x+1/2, -y, z+1/2). \tag{230b}$$

The $\mathbb{Z}_2$ cohomology ring is given by

$$\mathbb{Z}_2[A_c, A_m, A_y, B_\beta]/\langle \mathcal{R}_2, \mathcal{R}_3, \mathcal{R}_4\rangle \tag{231}$$

where the relations are

$$\mathcal{R}_2\colon \quad A_c(A_c + A_m), \quad A_m(A_c + A_m), \quad A_y(A_c + A_m + A_y), \tag{232a}$$
$$\mathcal{R}_3\colon \quad (A_c + A_m)B_\beta, \tag{232b}$$
$$\mathcal{R}_4\colon \quad B_\beta^2. \tag{232c}$$

We have the following table regarding IWPs and group cohomology at degree 3.

| Wyckoff position | Little group Intl. | Schönflies | Coordinates | LSM anomaly class | Topo. inv. |
|---|---|---|---|---|---|
| 2a | $m$ | $C_s$ | $(0,y,z)$, $(1/2, -y, z+1/2)$ | $A_y B_\beta$ | $\varphi_3[T_2, T_3, T_3^{-1}S_2 G]$ |

**No. 32: $Pba2$**

This group is generated by three translations $T_{1,2,3}$ as given in Eqs. (88), a two-fold rotation $C_2$, and a glide $G$:

$$C_2\colon (x,y,z) \to (-x, -y, z), \tag{233a}$$
$$G\colon (x,y,z) \to (x+1/2, -y+1/2, z). \tag{233b}$$

The $\mathbb{Z}_2$ cohomology ring is given by

$$\mathbb{Z}_2[A_c, A_m, A_z, B_\beta]/\langle \mathcal{R}_2, \mathcal{R}_3, \mathcal{R}_4\rangle \tag{234}$$

where the relations are

$$\mathcal{R}_2\colon \quad A_c A_m, \quad A_m^2, \quad A_z^2, \tag{235a}$$
$$\mathcal{R}_3\colon \quad A_m B_\beta, \tag{235b}$$
$$\mathcal{R}_4\colon \quad B_\beta(A_c^2 + B_\beta). \tag{235c}$$

We have the following table regarding IWPs and group cohomology at degree 3.

| Wyckoff position | Little group Intl. | Schönflies | Coordinates | LSM anomaly class | Topo. inv. |
|---|---|---|---|---|---|
| 2a | 2 | $C_2$ | $(0,0,z)$, $(1/2, 1/2, z)$ | $A_z(A_c^2 + B_\beta)$ | $\varphi_2[T_3, C_2]$ |
| 2b | 2 | $C_2$ | $(0,1/2,z)$, $(1/2, 0, z)$ | $A_z B_\beta$ | $\varphi_2[T_3, T_1 C_2]$ |

**No. 33: $Pna2_1$**

This group is generated by three translations $T_{1,2,3}$ as given in Eqs. (88), a two-fold screw $S_2$, and a glide $G$:

$$S_2\colon (x,y,z) \to (-x, -y, z+1/2), \tag{236a}$$
$$G\colon (x,y,z) \to (x+1/2, -y+1/2, z). \tag{236b}$$

The $\mathbb{Z}_2$ cohomology ring is given by

$$\mathbb{Z}_2[A_c, A_m, B_{\beta 1}, B_{\beta 2}]/\langle \mathcal{R}_2, \mathcal{R}_3, \mathcal{R}_4\rangle \tag{237}$$

where the relations are

$$\mathcal{R}_2: \quad A_c A_m, \quad A_c^2, \quad A_m^2, \tag{238a}$$

$$\mathcal{R}_3: \quad A_c B_{\beta 1}, \quad A_m B_{\beta 1} + A_c B_{\beta 2}, \quad A_m(B_{\beta 1} + B_{\beta 2}), \tag{238b}$$

$$\mathcal{R}_4: \quad B_{\beta 1}^2, \quad B_{\beta 1} B_{\beta 2}, \quad B_{\beta 2}^2. \tag{238c}$$

We have the following table regarding IWPs and group cohomology at degree 3.

| Wyckoff position | Little group Intl. | Little group Schönflies | Coordinates | LSM anomaly class | Topo. inv. |
|---|---|---|---|---|---|
| 4a | 1 | $C_1$ | $(x,y,z),\ (-x,-y,z+1/2),$ $(x+1/2,-y+1/2,z),\ (-x+1/2,y+1/2,z+1/2)$ | $A_m B_{\beta 1}$ | $\widehat{\varphi_4}[T_1, T_2, T_3^{-1} S_2, G]$ |

Here the topological invariant can be chosen to be

$$
\begin{aligned}
\widehat{\varphi_4}[T_1, T_2, T_3^{-1} S_2, G] =\ & \lambda((T_2, T_1, T_1^{-1}) + \lambda(T_1, T_2, T_1^{-1}) + \lambda(T_1, T_1^{-1}, T_2) + \lambda(T_2, 1, T_2) \\
& + \lambda(T_2, T_1^{-1} T_2^{-1} G_1, T_1^{-1} T_2^{-1} G_1) + \lambda(T_1^{-1} G_1, T_2, T_1^{-1} T_2^{-1} G_1) + \lambda(T_1^{-1} G_1, T_1^{-1} G_1, T_2) \\
& + \lambda(T_1, T_2, T_3^{-1} S_3 G_1) + \lambda(T_2, T_1, T_3^{-1} S_3 G_1) + \lambda(T_1, T_3^{-1} S_3 G_1, T_2) + \lambda(T_2, T_3^{-1} S_2 G_1, T_2) \\
& + \lambda(T_3^{-1} S_3, T_1^{-1} G_1, T_2) + \lambda(T_1^{-1} G_1, T_3^{-1} S_3, T_2) + \lambda(T_3^{-1} S_3, T_2, T_1^{-1} T_2^{-1} G_1)) \\
& + \lambda(T_1^{-1} G_1, T_2, T_2^{-1} T_3^{-1} S_3) + \lambda(T_2, T_2^{-1} T_3^{-1} S_3, T_1^{-1} T_2^{-1} G_1) + \lambda(T_2, T_1^{-1} T_2^{-1} G_1, T_2^{-1} T_3^{-1} S_3).
\end{aligned}
\tag{239}
$$

### No. 34: $Pnn2$

This group is generated by three translations $T_{1,2,3}$ as given in Eqs. (88), a two-fold rotation $C_2$, and a glide $G$:

$$C_2: (x,y,z) \to (-x,-y,z), \tag{240a}$$

$$G: (x,y,z) \to (x+1/2, -y+1/2, z+1/2). \tag{240b}$$

The $\mathbb{Z}_2$ cohomology ring is given by

$$\mathbb{Z}_2[A_c, A_m, A_{x+y+z}]/\langle \mathcal{R}_2, \mathcal{R}_3, \mathcal{R}_4 \rangle \tag{241}$$

where the relations are

$$\mathcal{R}_2: \quad A_c A_m, \quad A_m^2, \tag{242a}$$

$$\mathcal{R}_3: \quad A_m A_{x+y+z}^2, \tag{242b}$$

$$\mathcal{R}_4: \quad A_{x+y+z}^2 (A_c + A_{x+y+z})^2. \tag{242c}$$

We have the following table regarding IWPs and group cohomology at degree 3.

| Wyckoff position | Little group Intl. | Little group Schönflies | Coordinates | LSM anomaly class | Topo. inv. |
|---|---|---|---|---|---|
| 2a | 2 | $C_2$ | $(0,0,z),\ (1/2,1/2,z+1/2)$ | $A_{x+y+z}(A_c + A_{x+y+z})^2$ | $\varphi_2[T_3, C_2]$ |
| 2b | 2 | $C_2$ | $(0,1/2,z),\ (1/2,0,z+1/2)$ | $A_{x+y+z}^2(A_c + A_{x+y+z})$ | $\varphi_2[T_3, T_1 C_2]$ |

### No. 35: $Cmm2$

This group is generated by three translations $T_{1,2,3}$ as given in Eqs. (89), a two-fold rotation $C_2$, and a mirror $M$:

$$C_2: (x,y,z) \to (-x,-y,z), \tag{243a}$$

$$M: (x,y,z) \to (x,-y,z). \tag{243b}$$

The $\mathbb{Z}_2$ cohomology ring is given by

$$\mathbb{Z}_2[A_c, A_m, A_{x+y}, A_z, B_{xy}]/\langle \mathcal{R}_2, \mathcal{R}_3, \mathcal{R}_4 \rangle \tag{244}$$

where the relations are

$$\mathcal{R}_2: \quad A_m A_{x+y}, \quad A_{x+y}(A_c + A_{x+y}), \quad A_z^2, \tag{245a}$$

$$\mathcal{R}_3: \quad A_{x+y} B_{xy}, \tag{245b}$$

$$\mathcal{R}_4: \quad B_{xy}(A_c^2 + A_c A_m + B_{xy}). \tag{245c}$$

We have the following table regarding IWPs and group cohomology at degree 3.

| Wyckoff position | Little group Intl. | Little group Schönflies | Coordinates $(0,0,0) + (1/2,1/2,0)+$ | LSM anomaly class | Topo. inv. |
|---|---|---|---|---|---|
| 2a | $mm2$ | $C_{2v}$ | $(0,0,z)$ | $A_z(A_c^2 + A_c A_m + A_c A_{x+y} + B_{xy})$ | $\varphi_2[T_3, C_2]$ |
| 2b | $mm2$ | $C_{2v}$ | $(0,1/2,z)$ | $A_z B_{xy}$ | $\varphi_2[T_3, T_1 T_2 C_2]$ |
| 4c | $2$ | $C_2$ | $(1/4,1/4,z),\ (1/4,3/4,z)$ | $A_c A_{x+y} A_z$ | $\varphi_2[T_3, T_1 C_2]$ |

## No. 36: $Cmc2_1$

This group is generated by three translations $T_{1,2,3}$ as given in Eqs. (89), a two-fold screw $S_2$, and a glide $G$:

$$S_2: (x,y,z) \to (-x,-y,z+1/2), \tag{246a}$$

$$G: (x,y,z) \to (x,-y,z+1/2). \tag{246b}$$

The $\mathbb{Z}_2$ cohomology ring is given by

$$\mathbb{Z}_2[A_c, A_m, A_{x+y}, B_{xy}]/\langle \mathcal{R}_2, \mathcal{R}_3, \mathcal{R}_4 \rangle \tag{247}$$

where the relations are

$$\mathcal{R}_2: \quad A_m A_{x+y}, \quad (A_c + A_m)^2, \quad A_{x+y}(A_c + A_{x+y}), \tag{248a}$$

$$\mathcal{R}_3: \quad A_{x+y} B_{xy}, \tag{248b}$$

$$\mathcal{R}_4: \quad B_{xy}(A_c^2 + A_c A_m + B_{xy}). \tag{248c}$$

We have the following table regarding IWPs and group cohomology at degree 3.

| Wyckoff position | Little group Intl. | Little group Schönflies | Coordinates $(0,0,0) + (1/2,1/2,0)+$ | LSM anomaly class | Topo. inv. |
|---|---|---|---|---|---|
| 4a | $m$ | $C_s$ | $(0,y,z),\ (0,-y,z+1/2)$ | $(A_c + A_m)B_{xy}$ | $\widetilde{\varphi_3}[T_1 T_2, G, T_3^{-1} S_2 G]$ |

The expression of $\widetilde{\varphi_3}$ is given in Eq. (87).

## No. 37: $Ccc2$

This group is generated by three translations $T_{1,2,3}$ as given in Eqs. (89), a two-fold rotation $C_2$, and a glide $G$:

$$C_2: (x,y,z) \to (-x,-y,z), \tag{249a}$$

$$G: (x,y,z) \to (x,-y,z+1/2). \tag{249b}$$

The $\mathbb{Z}_2$ cohomology ring is given by

$$\mathbb{Z}_2[A_c, A_m, A_{x+y}, B_{xy}, B_{z(x+y)}]/\langle \mathcal{R}_2, \mathcal{R}_3, \mathcal{R}_4 \rangle \tag{250}$$

where the relations are

$$\mathcal{R}_2: \quad A_m A_{x+y}, \quad A_m^2, \quad A_{x+y}(A_c + A_{x+y}), \tag{251a}$$

$$\mathcal{R}_3: \quad A_{x+y} B_{xy}, \quad A_m B_{z(x+y)}, \quad A_m B_{xy} + A_c B_{z(x+y)} + A_{x+y} B_{z(x+y)}, \tag{251b}$$

$$\mathcal{R}_4: \quad B_{xy}(A_c^2 + A_c A_m + B_{xy}), \quad B_{xy}(A_c A_m + B_{z(x+y)}), \quad B_{z(x+y)}^2. \tag{251c}$$

We have the following table regarding IWPs and group cohomology at degree 3.

| Wyckoff position | Little group Intl. | Little group Schönflies | Coordinates $(0,0,0) + (1/2,1/2,0)+$ | LSM anomaly class | Topo. inv. |
|---|---|---|---|---|---|
| 4a | $2$ | $C_2$ | $(0,0,z),\ (0,0,z+1/2)$ | $A_m(A_c^2 + B_{xy})$ | $\varphi_2[G, C_2]$ |
| 4b | $2$ | $C_2$ | $(0,1/2,z),\ (0,1/2,z+1/2)$ | $A_m B_{xy}$ | $\varphi_2[G, T_1 T_2^{-1} C_2]$ |
| 4c | $2$ | $C_2$ | $(1/4,1/4,z),\ (1/4,3/4,z+1/2)$ | $A_m B_{xy} + A_c B_{z(x+y)}$ | $\varphi_2[T_3, T_1 C_2]$ |

**No. 38:** *Amm*2

This group is generated by three translations $T_{1,2,3}$ as given in Eqs. (90), a two-fold rotation $C_2$, and a mirror $M$:

$$C_2 \colon (x,y,z) \to (-x,-y,z), \tag{252a}$$
$$M \colon (x,y,z) \to (x,-y,z). \tag{252b}$$

The $\mathbb{Z}_2$ cohomology ring is given by

$$\mathbb{Z}_2[A_c, A_m, A_x, A_{y+z}, B_{yz}]/\langle \mathcal{R}_2, \mathcal{R}_3, \mathcal{R}_4 \rangle \tag{253}$$

where the relations are

$$\mathcal{R}_2 \colon \quad (A_c + A_m)A_{y+z}, \quad A_x(A_c + A_x), \quad A_{y+z}^2, \tag{254a}$$
$$\mathcal{R}_3 \colon \quad A_{y+z}B_{yz}, \tag{254b}$$
$$\mathcal{R}_4 \colon \quad B_{yz}^2. \tag{254c}$$

We have the following table regarding IWPs and group cohomology at degree 3.

| Wyckoff position | Little group Intl. | Little group Schönflies | Coordinates $(0,0,0)+$ $(0,1/2,1/2)+$ | LSM anomaly class | Topo. inv. |
|---|---|---|---|---|---|
| 2a | $mm2$ | $C_{2v}$ | $(0,0,z)$ | $(A_c + A_x)B_{yz}$ | $\varphi_2[T_2T_3, C_2]$ |
| 2b | $mm2$ | $C_{2v}$ | $(1/2,0,z)$ | $A_x B_{yz}$ | $\varphi_2[T_2T_3, T_1C_2]$ |

**No. 39:** *Aem*2

This group is generated by three translations $T_{1,2,3}$ as given in Eqs. (90), a two-fold rotation $C_2$, and a mirror $M$:

$$C_2 \colon (x,y,z) \to (-x,-y,z), \tag{255a}$$
$$M \colon (x,y,z) \to (x,-y+1/2,z). \tag{255b}$$

The $\mathbb{Z}_2$ cohomology ring is given by

$$\mathbb{Z}_2[A_c, A_m, A_x, A_{y+z}]/\langle \mathcal{R}_2 \rangle \tag{256}$$

where the relations are

$$\mathcal{R}_2 \colon \quad A_cA_m + A_cA_{y+z} + A_mA_{y+z}, \quad A_x(A_c + A_x), \quad A_{y+z}^2. \tag{257a}$$

We have the following table regarding IWPs and group cohomology at degree 3.

| Wyckoff position | Little group Intl. | Little group Schönflies | Coordinates $(0,0,0)+$ $(0,1/2,1/2)+$ | LSM anomaly class | Topo. inv. |
|---|---|---|---|---|---|
| 4a | 2 | $C_2$ | $(0,0,z), (0,1/2,z)$ | $A_c(A_cA_m + A_xA_{y+z})$ | $\varphi_2[T_3M, C_2]$ |
| 4b | 2 | $C_2$ | $(1/2,0,z), (1/2,1/2,z)$ | $A_cA_xA_{y+z}$ | $\varphi_2[T_3M, T_1C_2]$ |
| 4c | $m$ | $C_s$ | $(x,1/4,z), (-x,3/4,z)$ | $A_cA_x(A_m + A_{y+z})$ | $\widetilde{\varphi_3}[T_1, T_2C_2M, M]$ |

The expression of $\widetilde{\varphi_3}$ is given in Eq. (87).

**No. 40:** *Ama*2

This group is generated by three translations $T_{1,2,3}$ as given in Eqs. (90), a two-fold rotation $C_2$, and a glide $G$:

$$C_2 \colon (x,y,z) \to (-x,-y,z), \tag{258a}$$
$$G \colon (x,y,z) \to (x+1/2,-y,z). \tag{258b}$$

The $\mathbb{Z}_2$ cohomology ring is given by

$$\mathbb{Z}_2[A_c, A_m, A_{y+z}, B_{yz}, B_{x(y+z)}]/\langle \mathscr{R}_2, \mathscr{R}_3, \mathscr{R}_4 \rangle \tag{259}$$

where the relations are

$$\mathscr{R}_2: \quad (A_c + A_m)A_{y+z}, \quad A_m(A_c + A_m), \quad A_{y+z}^2, \tag{260a}$$

$$\mathscr{R}_3: \quad A_{y+z}B_{yz}, \quad (A_c + A_m)B_{x(y+z)}, \quad A_{y+z}B_{x(y+z)} + A_m B_{yz}, \tag{260b}$$

$$\mathscr{R}_4: \quad B_{yz}^2, \quad B_{x(y+z)}B_{yz}, \quad B_{x(y+z)}^2. \tag{260c}$$

We have the following table regarding IWPs and group cohomology at degree 3.

| Wyckoff position | Little group Intl. | Schönflies | Coordinates $(0,0,0)+$ $(0,1/2,1/2)+$ | LSM anomaly class | Topo. inv. |
|---|---|---|---|---|---|
| 4a | 2 | $C_2$ | $(0,0,z)$, $(1/2,0,z)$ | $(A_c + A_m)B_{yz}$ | $\varphi_2[T_2T_3, C_2]$ |
| 4b | $m$ | $C_s$ | $(1/4,y,z)$, $(3/4,-y,z)$ | $A_m B_{yz}$ | $\varphi_3[T_2, T_3, C_2 G]$ |

## No. 41: $Aea2$

This group is generated by three translations $T_{1,2,3}$ as given in Eqs. (90), a two-fold rotation $C_2$, and a glide $G$:

$$C_2: (x,y,z) \rightarrow (-x,-y,z), \tag{261a}$$

$$G: (x,y,z) \rightarrow (x+1/2, -y+1/2, z). \tag{261b}$$

The $\mathbb{Z}_2$ cohomology ring is given by

$$\mathbb{Z}_2[A_c, A_m, A_{y+z}, C_\gamma]/\langle \mathscr{R}_2, \mathscr{R}_4, \mathscr{R}_6 \rangle \tag{262}$$

where the relations are

$$\mathscr{R}_2: \quad A_c A_m + A_c A_{y+z} + A_m A_{y+z}, \quad A_m(A_c + A_m), \quad A_{y+z}^2, \tag{263a}$$

$$\mathscr{R}_4: \quad A_m C_\gamma, \quad A_{y+z}C_\gamma, \tag{263b}$$

$$\mathscr{R}_6: \quad C_\gamma^2. \tag{263c}$$

We have the following table regarding IWPs and group cohomology at degree 3.

| Wyckoff position | Little group Intl. | Schönflies | Coordinates $(0,0,0)+$ $(0,1/2,1/2)+$ | LSM anomaly class | Topo. inv. |
|---|---|---|---|---|---|
| 4a | 2 | $C_2$ | $(0,0,z)$, $(1/2,1/2,z)$ | $C_\gamma$ | $\varphi_2[T_2T_3, C_2]$ |

## No. 42: $Fmm2$

This group is generated by three translations $T_{1,2,3}$ as given in Eqs. (92), a two-fold rotation $C_2$, and a mirror $M$:

$$C_2: (x,y,z) \rightarrow (-x,-y,z), \tag{264a}$$

$$M: (x,y,z) \rightarrow (x,-y,z). \tag{264b}$$

The $\mathbb{Z}_2$ cohomology ring is given by

$$\mathbb{Z}_2[A_c, A_m, A_{x+z}, A_{y+z}, C_{xyz}]/\langle \mathscr{R}_2, \mathscr{R}_4, \mathscr{R}_6 \rangle \tag{265}$$

where the relations are

$$\mathscr{R}_2: \quad A_c A_{x+z} + A_m A_{x+z} + A_c A_{y+z}, \quad A_{x+z}^2 + A_c A_{y+z}, \quad A_{y+z}(A_c + A_{y+z}), \tag{266a}$$

$$\mathscr{R}_4: \quad A_{x+z}C_{xyz}, \quad A_{y+z}C_{xyz}, \tag{266b}$$

$$\mathscr{R}_6: \quad C_{xyz}^2. \tag{266c}$$

We have the following table regarding IWPs and group cohomology at degree 3.

| Wyckoff position | Little group Intl. | Schönflies | Coordinates $(0,0,0)+$ $(0,1/2,1/2)+$ $(1/2,0,1/2)+$ $(1/2,1/2,0)+$ | LSM anomaly class | Topo. inv. |
|---|---|---|---|---|---|
| 4a | $mm2$ | $C_{2v}$ | $(0,0,z)$ | $C_{xyz}$ | $\varphi_2[T_1T_2T_3^{-1}, C_2]$ |
| 8b | 2 | $C_2$ | $(1/4,1/4,z)$, $(1/4,3/4,z)$ | $A_c A_m A_{y+z}$ | $\varphi_2[T_1M, T_3C_2]$ |

**No. 43:** $Fdd2$

This group is generated by three translations $T_{1,2,3}$ as given in Eqs. (92), a two-fold rotation $C_2$, and a glide $G$:

$$C_2\colon (x,y,z) \to (-x,-y,z), \tag{267a}$$
$$G\colon (x,y,z) \to (x+1/4,-y+1/4,z+1/4). \tag{267b}$$

The $\mathbb{Z}_2$ cohomology ring is given by

$$\mathbb{Z}_2[A_c, A_m, B_{xy+xz+yz}, C_\gamma]/\langle \mathcal{R}_2, \mathcal{R}_3, \mathcal{R}_4, \mathcal{R}_5, \mathcal{R}_6 \rangle \tag{268}$$

where the relations are

$$\mathcal{R}_2\colon \quad A_c A_m, \quad A_m^2, \tag{269a}$$
$$\mathcal{R}_3\colon \quad A_c B_{xy+xz+yz}, \quad A_m B_{xy+xz+yz}, \tag{269b}$$
$$\mathcal{R}_4\colon \quad A_m C_\gamma, \quad B_{xy+xz+yz}^2, \tag{269c}$$
$$\mathcal{R}_5\colon \quad B_{xy+xz+yz} C_\gamma, \tag{269d}$$
$$\mathcal{R}_6\colon \quad C_\gamma^2. \tag{269e}$$

We have the following table regarding IWPs and group cohomology at degree 3.

| Wyckoff position | Little group Intl. | Schönflies | Coordinates $(0,0,0)+$ $(0,1/2,1/2)+$ $(1/2,0,1/2)+$ $(1/2,1/2,0)+$ | LSM anomaly class | Topo. inv. |
|---|---|---|---|---|---|
| 8a | 2 | $C_2$ | $(0,0,z), (1/4,1/4,z+1/4)$ | $C_\gamma$ | $\varphi_2[T_1 T_2 T_3^{-1}, C_2]$ |

**No. 44:** $Imm2$

This group is generated by three translations $T_{1,2,3}$ as given in Eqs. (91), a two-fold rotation $C_2$, and a mirror $M$:

$$C_2\colon (x,y,z) \to (-x,-y,z), \tag{270a}$$
$$M\colon (x,y,z) \to (x,-y,z). \tag{270b}$$

The $\mathbb{Z}_2$ cohomology ring is given by

$$\mathbb{Z}_2[A_c, A_m, A_{x+y+z}, B_\beta, B_{x(y+z)}, B_{y(x+z)}, C_{xyz}]/\langle \mathcal{R}_2, \mathcal{R}_3, \mathcal{R}_4, \mathcal{R}_5, \mathcal{R}_6 \rangle \tag{271}$$

where the relations are

$$\mathcal{R}_2\colon \quad A_c A_{x+y+z}, \quad A_m A_{x+y+z}, \quad A_{x+y+z}^2, \tag{272a}$$
$$\mathcal{R}_3\colon \quad A_{x+y+z} B_\beta, \quad A_{x+y+z} B_{x(y+z)}, \quad A_c B_\beta + A_m B_\beta + A_c B_{x(y+z)} + A_c B_{y(x+z)} + A_m B_{y(x+z)}, \quad A_{x+y+z} B_{y(x+z)}, \tag{272b}$$
$$\mathcal{R}_4\colon \quad A_{x+y+z} C_{xyz}, \quad A_c^2 B_\beta + A_c A_m B_\beta + B_\beta^2 + A_c A_m B_{x(y+z)}, \quad B_\beta B_{x(y+z)} + A_c C_{xyz} + A_m C_{xyz},$$
$$A_c^2 B_\beta + A_c A_m B_\beta + A_c A_m B_{x(y+z)} + B_\beta B_{y(x+z)} + A_c C_{xyz}, \quad B_{x(y+z)}^2, \quad B_{x(y+z)} B_{y(x+z)} + A_c C_{xyz} + A_m C_{xyz},$$
$$A_c^2 B_\beta + A_c A_m B_\beta + A_c A_m B_{x(y+z)} + B_{y(x+z)}^2, \tag{272c}$$
$$\mathcal{R}_5\colon \quad (A_c^2 + A_c A_m + B_\beta) C_{xyz}, \quad B_{x(y+z)} C_{xyz}, \quad (A_c^2 + A_c A_m + B_{y(x+z)}) C_{xyz}, \tag{272d}$$
$$\mathcal{R}_6\colon \quad C_{xyz}^2. \tag{272e}$$

We have the following table regarding IWPs and group cohomology at degree 3.

| Wyckoff position | Little group Intl. | Schönflies | Coordinates $(0,0,0)+$ $(1/2,1/2,1/2)+$ | LSM anomaly class | Topo. inv. |
|---|---|---|---|---|---|
| 2a | $mm2$ | $C_{2v}$ | $(0,0,z)$ | $A_c B_{x(y+z)} + C_{xyz}$ | $\varphi_2[T_1 T_2, C_2]$ |
| 2b | $mm2$ | $C_{2v}$ | $(0,1/2,z)$ | $C_{xyz}$ | $\varphi_2[T_1 T_2, T_1 T_3 C_2]$ |

**No. 45:** *Iba2*

This group is generated by three translations $T_{1,2,3}$ as given in Eqs. (91), a two-fold rotation $C_2$, and a glide $G$:

$$C_2 \colon (x,y,z) \to (-x,-y,z), \tag{273a}$$
$$G \colon (x,y,z) \to (x,-y,z+1/2). \tag{273b}$$

The $\mathbb{Z}_2$ cohomology ring is given by

$$\mathbb{Z}_2[A_c, A_m, A_{x+y+z}, B_{z(x+y)}]/\langle \mathcal{R}_2, \mathcal{R}_3, \mathcal{R}_4 \rangle \tag{274}$$

where the relations are

$$\mathcal{R}_2 \colon \quad A_m A_{x+y+z}, \quad A_m^2 + A_c A_{x+y+z}, \quad A_{x+y+z}(A_c + A_{x+y+z}), \tag{275a}$$
$$\mathcal{R}_3 \colon \quad A_{x+y+z} B_{z(x+y)}, \tag{275b}$$
$$\mathcal{R}_4 \colon \quad B_{z(x+y)}(A_c^2 + A_c A_m + B_{z(x+y)}). \tag{275c}$$

We have the following table regarding IWPs and group cohomology at degree 3.

| Wyckoff position | Little group Intl. | Schönflies | Coordinates $(0,0,0) +$ $(1/2,1/2,1/2)+$ | LSM anomaly class | Topo. inv. |
|---|---|---|---|---|---|
| 4a | 2 | $C_2$ | $(0,0,z)$, $(1/2,1/2,z)$ | $A_m(A_c^2 + B_{z(x+y)})$ | $\varphi_2[G, C_2]$ |
| 4b | 2 | $C_2$ | $(0,1/2,z)$, $(1/2,0,z)$ | $A_m B_{z(x+y)}$ | $\varphi_2[G, T_2 T_3 C_2]$ |

**No. 46:** *Ima2*

This group is generated by three translations $T_{1,2,3}$ as given in Eqs. (91), a two-fold rotation $C_2$, and a glide $G$:

$$C_2 \colon (x,y,z) \to (-x,-y,z), \tag{276a}$$
$$G \colon (x,y,z) \to (x+1/2,-y,z). \tag{276b}$$

The $\mathbb{Z}_2$ cohomology ring is given by

$$\mathbb{Z}_2[A_c, A_m, A_{x+y+z}, B_{x(y+z)}, B_{y(x+z)}]/\langle \mathcal{R}_2, \mathcal{R}_3, \mathcal{R}_4 \rangle \tag{277}$$

where the relations are

$$\mathcal{R}_2 \colon \quad (A_c + A_m)A_{x+y+z}, \quad A_c A_m + A_m^2 + A_c A_{x+y+z}, \quad A_{x+y+z}^2, \tag{278a}$$
$$\mathcal{R}_3 \colon \quad A_{x+y+z} B_{x(y+z)}, \quad A_c^3 + A_c A_m^2 + A_c B_{x(y+z)} + A_c B_{y(x+z)} + A_m B_{y(x+z)}, \quad A_m B_{x(y+z)} + A_{x+y+z} B_{y(x+z)}, \tag{278b}$$
$$\mathcal{R}_4 \colon \quad B_{x(y+z)}^2, \quad B_{x(y+z)}(A_c^2 + A_c A_m + B_{y(x+z)}), \quad A_c^4 + A_c^2 A_m^2 + A_c A_m B_{x(y+z)} + B_{y(x+z)}^2. \tag{278c}$$

We have the following table regarding IWPs and group cohomology at degree 3.

| Wyckoff position | Little group Intl. | Schönflies | Coordinates $(0,0,0) +$ $(1/2,1/2,1/2)+$ | LSM anomaly class | Topo. inv. |
|---|---|---|---|---|---|
| 4a | 2 | $C_2$ | $(0,0,z)$, $(1/2,0,z)$ | $(A_c + A_m)B_{x(y+z)}$ | $\varphi_2[T_1 T_2, C_2]$ |
| 4b | m | $C_s$ | $(1/4,y,z)$, $(3/4,-y,z)$ | $A_m B_{x(y+z)}$ | $\widetilde{\varphi_3}[T_1 T_3, T_1 G, C_2 G]$ |

The expression of $\widetilde{\varphi_3}$ is given in Eq. (87).

**No. 47:** *Pmmm*

This group is generated by three translations $T_{1,2,3}$ as given in Eqs. (88), a two-fold rotation $C_2$, a two-fold rotation $C_2'$, and an inversion $I$:

$$C_2 \colon (x,y,z) \to (-x,-y,z), \tag{279a}$$
$$C_2' \colon (x,y,z) \to (-x,y,-z), \tag{279b}$$
$$I \colon (x,y,z) \to (-x,-y,-z). \tag{279c}$$

The $\mathbb{Z}_2$ cohomology ring is given by

$$\mathbb{Z}_2[A_i, A_{c'}, A_c, A_x, A_y, A_z]/\langle \mathscr{R}_2 \rangle \tag{280}$$

where the relations are

$$\mathscr{R}_2: \quad A_x(A_c + A_{c'} + A_i + A_x), \quad A_y(A_c + A_i + A_y), \quad A_z(A_{c'} + A_i + A_z). \tag{281a}$$

We have the following table regarding IWPs and group cohomology at degree 3.

| Wyckoff position | Little group Intl. | Little group Schönflies | Coordinates | LSM anomaly class | Topo. inv. |
|---|---|---|---|---|---|
| 1a | $mmm$ | $D_{2h}$ | $(0,0,0)$ | $(A_c + A_{c'} + A_i + A_x)(A_c + A_i + A_y)(A_{c'} + A_i + A_z)$ | $\varphi_2[C_2, C_2']$ |
| 1b | $mmm$ | $D_{2h}$ | $(1/2,0,0)$ | $A_x(A_c + A_i + A_y)(A_{c'} + A_i + A_z)$ | $\varphi_2[T_1C_2, T_1C_2']$ |
| 1c | $mmm$ | $D_{2h}$ | $(0,0,1/2)$ | $(A_c + A_{c'} + A_i + A_x)(A_c + A_i + A_y)A_z$ | $\varphi_2[C_2, T_3C_2']$ |
| 1d | $mmm$ | $D_{2h}$ | $(1/2,0,1/2)$ | $A_x(A_c + A_i + A_y)A_z$ | $\varphi_2[T_1C_2, T_1T_3C_2']$ |
| 1e | $mmm$ | $D_{2h}$ | $(0,1/2,0)$ | $(A_c + A_{c'} + A_i + A_x)A_y(A_{c'} + A_i + A_z)$ | $\varphi_2[T_2C_2, C_2']$ |
| 1f | $mmm$ | $D_{2h}$ | $(1/2,1/2,0)$ | $A_xA_y(A_{c'} + A_i + A_z)$ | $\varphi_2[T_1T_2C_2, T_1C_2']$ |
| 1g | $mmm$ | $D_{2h}$ | $(0,1/2,1/2)$ | $(A_c + A_{c'} + A_i + A_x)A_yA_z$ | $\varphi_2[T_2C_2, T_3C_2']$ |
| 1h | $mmm$ | $D_{2h}$ | $(1/2,1/2,1/2)$ | $A_xA_yA_z$ | $\varphi_2[T_1T_2C_2, T_1T_3C_2']$ |

## No. 48: $Pnnn$

This group is generated by three translations $T_{1,2,3}$ as given in Eqs. (88), a two-fold rotation $C_2$, a two-fold rotation $C_2'$, and an inversion $I$:

$$C_2: (x,y,z) \to (-x+1/2, -y+1/2, z), \tag{282a}$$
$$C_2': (x,y,z) \to (-x+1/2, y, -z+1/2), \tag{282b}$$
$$I: (x,y,z) \to (-x,-y,-z). \tag{282c}$$

The $\mathbb{Z}_2$ cohomology ring is given by

$$\mathbb{Z}_2[A_i, A_{c'}, A_c, A_{x+y+z}]/\langle \mathscr{R}_2, \mathscr{R}_3, \mathscr{R}_4 \rangle \tag{283}$$

where the relations are

$$\mathscr{R}_2: \quad A_{c'}A_i, \quad A_cA_i, \tag{284a}$$
$$\mathscr{R}_3: \quad A_iA_{x+y+z}(A_i + A_{x+y+z}), \tag{284b}$$
$$\mathscr{R}_4: \quad A_{x+y+z}(A_c^2A_{c'} + A_cA_{c'}^2 + A_i^3 + A_c^2A_{x+y+z} + A_cA_{c'}A_{x+y+z} + A_{c'}^2A_{x+y+z} + A_{x+y+z}^3). \tag{284c}$$

We have the following table regarding IWPs and group cohomology at degree 3.

| Wyckoff position | Little group Intl. | Little group Schönflies | Coordinates | LSM anomaly class | Topo. inv. |
|---|---|---|---|---|---|
| 2a | 222 | $D_2$ | $(1/4,1/4,1/4), (3/4,3/4,3/4)$ | $A_c^2A_{c'} + A_cA_{c'}^2 + A_c^2A_{x+y+z} + A_cA_{c'}A_{x+y+z} + A_{c'}^2A_{x+y+z} + A_i^2A_{x+y+z} + A_{x+y+z}^3$ | $\varphi_2[C_2, C_2']$ |
| 2b | 222 | $D_2$ | $(3/4,1/4,1/4), (1/4,3/4,3/4)$ | $A_{x+y+z}(A_cA_{c'} + A_i^2 + A_cA_{x+y+z} + A_{c'}A_{x+y+z} + A_{x+y+z}^2)$ | $\varphi_2[T_1C_2, T_1C_2']$ |
| 2c | 222 | $D_2$ | $(1/4,1/4,3/4), (3/4,3/4,1/4)$ | $A_{x+y+z}(A_c^2 + A_cA_{c'} + A_i^2 + A_{c'}A_{x+y+z} + A_{x+y+z}^2)$ | $\varphi_2[C_2, T_3C_2']$ |
| 2d | 222 | $D_2$ | $(1/4,3/4,1/4), (3/4,1/4,3/4)$ | $A_{x+y+z}(A_cA_{c'} + A_{c'}^2 + A_i^2 + A_cA_{x+y+z} + A_{x+y+z}^2)$ | $\varphi_2[T_1C_2, T_1T_3C_2']$ |
| 4e | $\bar{1}$ | $C_i$ | $(1/2,1/2,1/2), (0,0,1/2),$ $(0,1/2,0), (1/2,0,0)$ | $A_i^2A_{x+y+z}$ | $\varphi_1[T_1T_2T_3I]$ |
| 4f | $\bar{1}$ | $C_i$ | $(0,0,0), (1/2,1/2,0),$ $(1/2,0,1/2), (0,1/2,1/2)$ | $A_i^2(A_i + A_{x+y+z})$ | $\varphi_1[I]$ |

## No. 49: $Pccm$

This group is generated by three translations $T_{1,2,3}$ as given in Eqs. (88), a two-fold rotation $C_2$, a two-fold rotation $C_2'$, and an inversion $I$:

$$C_2: (x,y,z) \to (-x,-y,z), \tag{285a}$$
$$C_2': (x,y,z) \to (-x,y,-z+1/2), \tag{285b}$$
$$I: (x,y,z) \to (-x,-y,-z). \tag{285c}$$

The $\mathbb{Z}_2$ cohomology ring is given by

$$\mathbb{Z}_2[A_i, A_{c'}, A_c, A_x, A_y]/\langle \mathcal{R}_2 \rangle \tag{286}$$

where the relations are

$$\mathcal{R}_2: \quad A_{c'}A_i, \quad A_x(A_c + A_{c'} + A_i + A_x), \quad A_y(A_c + A_i + A_y). \tag{287a}$$

We have the following table regarding IWPs and group cohomology at degree 3.

| Wyckoff position | Little group | | Coordinates | LSM anomaly class | Topo. inv. |
|---|---|---|---|---|---|
| | Intl. | Schönflies | | | |
| 2a | $2/m$ | $C_{2h}$ | $(0,0,0)$, $(0,0,1/2)$ | $A_i(A_c + A_i + A_x)(A_c + A_i + A_y)$ | $\varphi_1[I]$ |
| 2b | $2/m$ | $C_{2h}$ | $(1/2,1/2,0)$, $(1/2,1/2,1/2)$ | $A_i A_x A_y$ | $\varphi_1[T_1 T_2 I]$ |
| 2c | $2/m$ | $C_{2h}$ | $(0,1/2,0)$, $(0,1/2,1/2)$ | $A_i(A_c + A_i + A_x)A_y$ | $\varphi_1[T_2 I]$ |
| 2d | $2/m$ | $C_{2h}$ | $(1/2,0,0)$, $(1/2,0,1/2)$ | $A_i A_x(A_c + A_i + A_y)$ | $\varphi_1[T_1 I]$ |
| 2e | $222$ | $D_2$ | $(0,0,1/4)$, $(0,0,3/4)$ | $A_{c'}(A_c + A_{c'} + A_x)(A_c + A_y)$ | $\varphi_2[C_2, C_2']$ |
| 2f | $222$ | $D_2$ | $(1/2,0,1/4)$, $(1/2,0,3/4)$ | $A_{c'}A_x(A_c + A_y)$ | $\varphi_2[T_1 C_2, T_1 C_2']$ |
| 2g | $222$ | $D_2$ | $(0,1/2,1/4)$, $(0,1/2,3/4)$ | $A_{c'}(A_c + A_{c'} + A_x)A_y$ | $\varphi_2[T_2 C_2, C_2']$ |
| 2h | $222$ | $D_2$ | $(1/2,1/2,1/4)$, $(1/2,1/2,3/4)$ | $A_{c'}A_x A_y$ | $\varphi_2[T_1 T_2 C_2, T_1 C_2']$ |

## No. 50: *Pban*

This group is generated by three translations $T_{1,2,3}$ as given in Eqs. (88), a two-fold rotation $C_2$, a two-fold rotation $C_2'$, and an inversion $I$:

$$C_2: (x,y,z) \rightarrow (-x+1/2, -y+1/2, z), \tag{288a}$$
$$C_2': (x,y,z) \rightarrow (-x+1/2, y, -z), \tag{288b}$$
$$I: (x,y,z) \rightarrow (-x, -y, -z). \tag{288c}$$

The $\mathbb{Z}_2$ cohomology ring is given by

$$\mathbb{Z}_2[A_i, A_{c'}, A_c, A_z, B_\beta]/\langle \mathcal{R}_2, \mathcal{R}_3, \mathcal{R}_4 \rangle \tag{289}$$

where the relations are

$$\mathcal{R}_2: \quad A_{c'}A_i, \quad A_c A_i, \quad A_z(A_{c'} + A_i + A_z), \tag{290a}$$
$$\mathcal{R}_3: \quad A_i B_\beta, \tag{290b}$$
$$\mathcal{R}_4: \quad B_\beta(A_c^2 + A_c A_{c'} + B_\beta). \tag{290c}$$

We have the following table regarding IWPs and group cohomology at degree 3.

| Wyckoff position | Little group | | Coordinates | LSM anomaly class | Topo. inv. |
|---|---|---|---|---|---|
| | Intl. | Schönflies | | | |
| 2a | $222$ | $D_2$ | $(1/4,1/4,0)$, $(3/4,3/4,0)$ | $(A_{c'} + A_z)(A_c^2 + A_c A_{c'} + B_\beta)$ | $\varphi_2[C_2, C_2']$ |
| 2b | $222$ | $D_2$ | $(3/4,1/4,0)$, $(1/4,3/4,0)$ | $(A_{c'} + A_z)B_\beta$ | $\varphi_2[T_2 C_2, C_2']$ |
| 2c | $222$ | $D_2$ | $(3/4,1/4,1/2)$, $(1/4,3/4,1/2)$ | $A_z B_\beta$ | $\varphi_2[T_2 C_2, T_3 C_2']$ |
| 2d | $222$ | $D_2$ | $(1/4,1/4,1/2)$, $(3/4,3/4,1/2)$ | $A_z(A_c^2 + A_c A_{c'} + B_\beta)$ | $\varphi_2[C_2, T_3 C_2']$ |
| 4e | $\bar{1}$ | $C_i$ | $(0,0,0)$, $(1/2,1/2,0)$, $(1/2,0,0)$, $(0,1/2,0)$ | $A_i^2(A_i + A_z)$ | $\varphi_1[I]$ |
| 4f | $\bar{1}$ | $C_i$ | $(0,0,1/2)$, $(1/2,1/2,1/2)$, $(1/2,0,1/2)$, $(0,1/2,1/2)$ | $A_i^2 A_z$ | $\varphi_1[T_3 I]$ |

## No. 51: *Pmma*

This group is generated by three translations $T_{1,2,3}$ as given in Eqs. (88), a two-fold rotation $C_2$, a two-fold rotation $C_2'$, and an inversion $I$:

$$C_2: (x,y,z) \rightarrow (-x+1/2, -y, z), \tag{291a}$$
$$C_2': (x,y,z) \rightarrow (-x, y, -z), \tag{291b}$$
$$I: (x,y,z) \rightarrow (-x, -y, -z). \tag{291c}$$

The $\mathbb{Z}_2$ cohomology ring is given by

$$\mathbb{Z}_2[A_i, A_{c'}, A_c, A_y, A_z]/\langle\mathcal{R}_2\rangle \tag{292}$$

where the relations are

$$\mathcal{R}_2: \quad A_c(A_{c'} + A_i), \quad A_y(A_c + A_i + A_y), \quad A_z(A_{c'} + A_i + A_z). \tag{293a}$$

We have the following table regarding IWPs and group cohomology at degree 3.

| Wyckoff position | Little group Intl. | Little group Schönflies | Coordinates | LSM anomaly class | Topo. inv. |
|---|---|---|---|---|---|
| 2a | $2m$ | $C_{2h}$ | $(0,0,0)$, $(1/2,0,0)$ | $(A_{c'} + A_i)(A_i + A_y)(A_{c'} + A_i + A_z)$ | $\varphi_1[I]$ |
| 2b | $2m$ | $C_{2h}$ | $(0,1/2,0)$, $(1/2,1/2,0)$ | $(A_{c'} + A_i)A_y(A_{c'} + A_i + A_z)$ | $\varphi_1[T_2I]$ |
| 2c | $2m$ | $C_{2h}$ | $(0,0,1/2)$, $(1/2,0,1/2)$ | $(A_{c'} + A_i)(A_i + A_y)A_z$ | $\varphi_1[T_3I]$ |
| 2d | $2m$ | $C_{2h}$ | $(0,1/2,1/2)$, $(1/2,1/2,1/2)$ | $(A_{c'} + A_i)A_yA_z$ | $\varphi_1[T_2T_3I]$ |
| 2e | $mm2$ | $C_{2v}$ | $(1/4,0,z)$, $(3/4,0,-z)$ | $A_c(A_c + A_i + A_y)A_z$ | $\varphi_2[T_3, C_2]$ |
| 2f | $mm2$ | $C_{2v}$ | $(1/4,1/2,z)$, $(3/4,1/2,-z)$ | $A_cA_yA_z$ | $\varphi_2[T_3, T_2C_2]$ |

## No. 52: $Pnna$

This group is generated by three translations $T_{1,2,3}$ as given in Eqs. (88), a two-fold rotation $C_2$, a two-fold screw $S_2'$, and an inversion $I$:

$$C_2: (x,y,z) \to (-x+1/2, -y, z), \tag{294a}$$
$$S_2': (x,y,z) \to (-x+1/2, y+1/2, -z+1/2), \tag{294b}$$
$$I: (x,y,z) \to (-x,-y,-z). \tag{294c}$$

The $\mathbb{Z}_2$ cohomology ring is given by

$$\mathbb{Z}_2[A_i, A_{c'}, A_c, B_{\beta1}, B_{\beta2}]/\langle\mathcal{R}_2, \mathcal{R}_3, \mathcal{R}_4\rangle \tag{295}$$

where the relations are

$$\mathcal{R}_2: \quad A_{c'}A_i, \quad A_cA_i, \quad A_{c'}(A_c + A_{c'}), \tag{296a}$$
$$\mathcal{R}_3: \quad A_{c'}B_{\beta1}, \quad A_i(B_{\beta1} + B_{\beta2}), \quad (A_c + A_{c'})B_{\beta2}, \tag{296b}$$
$$\mathcal{R}_4: \quad B_{\beta1}(A_i^2 + B_{\beta1}), \quad B_{\beta1}(A_i^2 + B_{\beta2}), \quad A_i^2B_{\beta1} + B_{\beta2}^2. \tag{296c}$$

We have the following table regarding IWPs and group cohomology at degree 3.

| Wyckoff position | Little group Intl. | Little group Schönflies | Coordinates | LSM anomaly class | Topo. inv. |
|---|---|---|---|---|---|
| 4a | $\bar{1}$ | $C_i$ | $(0,0,0)$, $(1/2,0,0)$, $(1/2,1/2,1/2)$, $(0,1/2,1/2)$ | $A_iB_{\beta1}$ | $\varphi_1[I]$ |
| 4b | $\bar{1}$ | $C_i$ | $(0,0,1/2)$, $(1/2,0,1/2)$, $(1/2,1/2,0)$, $(0,1/2,0)$ | $A_i(A_i^2 + B_{\beta1})$ | $\varphi_1[T_2I]$ |
| 4c | $2$ | $C_2$ | $(1/4,0,z)$, $(1/4,1/2,-z+1/2)$, $(3/4,0,-z)$, $(3/4,1/2,z+1/2)$ | $A_cB_{\beta1}$ | $\varphi_2[T_3, C_2]$ |
| 4d | $2$ | $C_2$ | $(x,1/4,1/4)$, $(-x+1/2,3/4,1/4)$, $(-x,3/4,3/4)$, $(x+1/2,1/4,3/4)$ | $A_{c'}B_{\beta2}$ | $\varphi_2[T_1, C_2S_2']$ |

## No. 53: $Pmna$

This group is generated by three translations $T_{1,2,3}$ as given in Eqs. (88), a two-fold screw $S_2$, a two-fold rotation $C_2'$, and an inversion $I$:

$$S_2: (x,y,z) \to (-x+1/2, -y, z+1/2), \tag{297a}$$
$$C_2': (x,y,z) \to (-x+1/2, y, -z+1/2), \tag{297b}$$
$$I: (x,y,z) \to (-x,-y,-z). \tag{297c}$$

The $\mathbb{Z}_2$ cohomology ring is given by

$$\mathbb{Z}_2[A_i, A_{c'}, A_c, A_y, B_\beta]/\langle \mathscr{R}_2, \mathscr{R}_3, \mathscr{R}_4 \rangle \tag{298}$$

where the relations are

$$\mathscr{R}_2: \quad (A_c + A_{c'})A_i, \quad A_c(A_c + A_{c'}), \quad A_y(A_c + A_i + A_y), \tag{299a}$$

$$\mathscr{R}_3: \quad (A_c + A_{c'})B_\beta, \tag{299b}$$

$$\mathscr{R}_4: \quad B_\beta(A_{c'}A_i + A_i^2 + B_\beta). \tag{299c}$$

We have the following table regarding IWPs and group cohomology at degree 3.

| Wyckoff position | Little group Intl. | Little group Schönflies | Coordinates | LSM anomaly class | Topo. inv. |
|---|---|---|---|---|---|
| 2a | $2/m$ | $C_{2h}$ | $(0,0,0)$, $(1/2,0,1/2)$ | $(A_{c'} + A_i + A_y)B_\beta$ | $\varphi_1[I]$ |
| 2b | $2/m$ | $C_{2h}$ | $(1/2,0,0)$, $(0,0,1/2)$ | $(A_{c'} + A_i + A_y)(A_{c'}A_i + A_i^2 + B_\beta)$ | $\varphi_1[T_1 I]$ |
| 2c | $2/m$ | $C_{2h}$ | $(1/2,1/2,0)$, $(0,1/2,1/2)$ | $A_y(A_{c'}A_i + A_i^2 + B_\beta)$ | $\varphi_1[T_1 T_2 I]$ |
| 2d | $2/m$ | $C_{2h}$ | $(0,1/2,0)$, $(1/2,1/2,1/2)$ | $A_y B_\beta$ | $\varphi_1[T_2 I]$ |
| 4g | 2 | $C_2$ | $(1/4,y,1/4)$, $(1/4,-y,3/4)$, $(3/4,-y,3/4)$, $(3/4,y,1/4)$ | $A_{c'}(A_c + A_{c'})A_y$ | $\varphi_2[T_2, C_2']$ |

### No. 54: *Pcca*

This group is generated by three translations $T_{1,2,3}$ as given in Eqs. (88), a two-fold rotation $C_2$, a two-fold rotation $C_2'$, and an inversion $I$:

$$C_2: (x,y,z) \rightarrow (-x+1/2, -y, z), \tag{300a}$$

$$C_2': (x,y,z) \rightarrow (-x, y, -z+1/2), \tag{300b}$$

$$I: (x,y,z) \rightarrow (-x, -y, -z). \tag{300c}$$

The $\mathbb{Z}_2$ cohomology ring is given by

$$\mathbb{Z}_2[A_i, A_{c'}, A_c, A_y]/\langle \mathscr{R}_2 \rangle \tag{301}$$

where the relations are

$$\mathscr{R}_2: \quad A_{c'}A_i, \quad A_c(A_{c'} + A_i), \quad A_y(A_c + A_i + A_y). \tag{302a}$$

We have the following table regarding IWPs and group cohomology at degree 3.

| Wyckoff position | Little group Intl. | Little group Schönflies | Coordinates | LSM anomaly class | Topo. inv. |
|---|---|---|---|---|---|
| 4a | $\bar{1}$ | $C_i$ | $(0,0,0)$, $(1/2,0,0)$, $(0,0,1/2)$, $(1/2,0,1/2)$ | $A_i^2(A_i + A_y)$ | $\varphi_1[I]$ |
| 4b | $\bar{1}$ | $C_i$ | $(0,1/2,0)$, $(1/2,1/2,0)$, $(0,1/2,1/2)$, $(1/2,1/2,1/2)$ | $A_i^2 A_y$ | $\varphi_1[T_2 I]$ |
| 4c | 2 | $C_2$ | $(0,y,1/4)$, $(1/2,-y,1/4)$, $(0,-y,3/4)$, $(1/2,y,3/4)$ | $A_{c'}^2 A_y$ | $\varphi_2[T_2, C_2']$ |
| 4d | 2 | $C_2$ | $(1/4,0,z)$, $(3/4,0,-z+1/2)$, $(3/4,0,-z)$, $(1/4,0,z+1/2)$ | $A_c A_i(A_c + A_y)$ | $\varphi_2[C_2'I, C_2]$ |
| 4e | 2 | $C_2$ | $(1/4,1/2,z)$, $(3/4,1/2,-z+1/2)$, $(3/4,1/2,-z)$, $(1/4,1/2,z+1/2)$ | $A_c A_i A_y$ | $\varphi_2[T_2 C_2'I, T_2 C_2]$ |

### No. 55: *Pbam*

This group is generated by three translations $T_{1,2,3}$ as given in Eqs. (88), a two-fold rotation $C_2$, a two-fold screw $S_2'$, and an inversion $I$:

$$C_2: (x,y,z) \rightarrow (-x, -y, z), \tag{303a}$$

$$S_2': (x,y,z) \rightarrow (-x+1/2, y+1/2, -z), \tag{303b}$$

$$I: (x,y,z) \rightarrow (-x, -y, -z). \tag{303c}$$

The $\mathbb{Z}_2$ cohomology ring is given by

$$\mathbb{Z}_2[A_i, A_{c'}, A_c, A_z, B_\beta]/\langle \mathcal{R}_2, \mathcal{R}_3, \mathcal{R}_4 \rangle \tag{304}$$

where the relations are

$$\mathcal{R}_2: \quad A_{c'}(A_c + A_i), \quad A_{c'}^2, \quad A_z(A_{c'} + A_i + A_z), \tag{305a}$$
$$\mathcal{R}_3: \quad A_{c'}B_\beta, \tag{305b}$$
$$\mathcal{R}_4: \quad B_\beta(A_c^2 + A_i^2 + B_\beta). \tag{305c}$$

We have the following table regarding IWPs and group cohomology at degree 3.

| Wyckoff position | Little group Intl. | Little group Schönflies | Coordinates | LSM anomaly class | Topo. inv. |
|---|---|---|---|---|---|
| 2a | $2/m$ | $C_{2h}$ | $(0,0,0)$, $(1/2,1/2,0)$ | $(A_i + A_z)(A_c^2 + A_i^2 + B_\beta)$ | $\varphi_1[I]$ |
| 2b | $2/m$ | $C_{2h}$ | $(0,0,1/2)$, $(1/2,1/2,1/2)$ | $A_z(A_c^2 + A_i^2 + B_\beta)$ | $\varphi_1[T_3I]$ |
| 2c | $2/m$ | $C_{2h}$ | $(0,1/2,0)$, $(1/2,0,0)$ | $(A_i + A_z)B_\beta$ | $\varphi_1[T_1I]$ |
| 2d | $2/m$ | $C_{2h}$ | $(0,1/2,1/2)$, $(1/2,0,1/2)$ | $A_zB_\beta$ | $\varphi_1[T_1T_3I]$ |

**No. 56:** *Pccn*

This group is generated by three translations $T_{1,2,3}$ as given in Eqs. (88), a two-fold rotation $C_2$, a two-fold screw $S_2'$, and an inversion $I$:

$$C_2: (x,y,z) \to (-x+1/2, -y+1/2, z), \tag{306a}$$
$$S_2': (x,y,z) \to (-x, y+1/2, -z+1/2), \tag{306b}$$
$$I: (x,y,z) \to (-x, -y, -z). \tag{306c}$$

The $\mathbb{Z}_2$ cohomology ring is given by

$$\mathbb{Z}_2[A_i, A_{c'}, A_c, B_{\beta 1}, B_{\beta 2}]/\langle \mathcal{R}_2, \mathcal{R}_3, \mathcal{R}_4 \rangle \tag{307}$$

where the relations are

$$\mathcal{R}_2: \quad A_{c'}A_i, \quad A_c(A_{c'} + A_i), \quad A_{c'}^2, \tag{308a}$$
$$\mathcal{R}_3: \quad (A_{c'} + A_i)B_{\beta 1}, \quad A_{c'}B_{\beta 2}, \quad A_iB_{\beta 1} + A_cB_{\beta 2}, \tag{308b}$$
$$\mathcal{R}_4: \quad (A_c^2 + B_{\beta 1})(A_cA_i + B_{\beta 1}), \quad B_{\beta 1}(A_cA_i + B_{\beta 2}), \quad B_{\beta 2}(A_i^2 + B_{\beta 2}). \tag{308c}$$

We have the following table regarding IWPs and group cohomology at degree 3.

| Wyckoff position | Little group Intl. | Little group Schönflies | Coordinates | LSM anomaly class | Topo. inv. |
|---|---|---|---|---|---|
| 4a | $\bar{1}$ | $C_i$ | $(0,0,0)$, $(1/2,1/2,0)$, $(0,1/2,1/2)$, $(1/2,0,1/2)$ | $A_i(A_i^2 + B_{\beta 2})$ | $\varphi_1[I]$ |
| 4b | $\bar{1}$ | $C_i$ | $(0,0,1/2)$, $(1/2,1/2,1/2)$, $(0,1/2,0)$, $(1/2,0,0)$ | $A_iB_{\beta 2}$ | $\varphi_1[T_1I]$ |
| 4c | $2$ | $C_2$ | $(1/4,1/4,z)$, $(3/4,3/4,-z+1/2)$, $(3/4,3/4,-z)$, $(1/4,1/4,z+1/2)$ | $A_i(A_c^2 + B_{\beta 1})$ | $\varphi_2[S_2'I, C_2]$ |
| 4d | $2$ | $C_2$ | $(1/4,3/4,z)$, $(3/4,1/4,-z+1/2)$, $(3/4,1/4,-z)$, $(1/4,3/4,z+1/2)$ | $A_iB_{\beta 1}$ | $\varphi_2[S_2'I, T_1C_2]$ |

**No. 57:** *Pbcm*

This group is generated by three translations $T_{1,2,3}$ as given in Eqs. (88), a two-fold screw $S_2$, a two-fold screw $S_2'$, and an inversion $I$:

$$S_2: (x,y,z) \to (-x, -y, z+1/2), \tag{309a}$$
$$S_2': (x,y,z) \to (-x, y+1/2, -z+1/2), \tag{309b}$$
$$I: (x,y,z) \to (-x, -y, -z). \tag{309c}$$

The $\mathbb{Z}_2$ cohomology ring is given by

$$\mathbb{Z}_2[A_i, A_{c'}, A_c, A_x]/\langle \mathscr{R}_2 \rangle \tag{310}$$

where the relations are

$$\mathscr{R}_2: \quad A_{c'}(A_c + A_{c'} + A_i), \quad (A_c + A_{c'})(A_c + A_i), \quad A_x(A_c + A_{c'} + A_i + A_x). \tag{311a}$$

We have the following table regarding IWPs and group cohomology at degree 3.

| Wyckoff position | Little group Intl. | Little group Schönflies | Coordinates | LSM anomaly class | Topo. inv. |
|---|---|---|---|---|---|
| 4a | $\bar{1}$ | $C_i$ | $(0,0,0),\ (0,0,1/2),$ $(0,1/2,1/2),\ (0,1/2,0)$ | $A_i(A_c + A_i)(A_i + A_x)$ | $\varphi_1[I]$ |
| 4b | $\bar{1}$ | $C_i$ | $(1/2,0,0),\ (1/2,0,1/2),$ $(1/2,1/2,1/2),\ (1/2,1/2,0)$ | $A_i(A_c + A_i)A_x$ | $\varphi_1[T_1 I]$ |
| 4c | $2$ | $C_2$ | $(x,1/4,0),\ (-x,3/4,1/2),$ $(-x,3/4,0),\ (x,1/4,1/2)$ | $A_{c'}(A_c + A_i)A_x$ | $\varphi_2[T_1, S_2 S_2']$ |
| 4d | $m$ | $C_s$ | $(x,y,1/4),\ (-x,-y,3/4),$ $(-x,y+1/2,1/4),\ (x,-y+1/2,3/4)$ | $A_{c'}A_i A_x$ | $\widetilde{\varphi_3}[T_1, T_3^{-1} S_2 S_2' I, S_2 I]$ |

The expression of $\widetilde{\varphi_3}$ is given in Eq. (87).

## No. 58: $Pnnm$

This group is generated by three translations $T_{1,2,3}$ as given in Eqs. (88), a two-fold rotation $C_2$, a two-fold screw $S_2'$, and an inversion $I$:

$$C_2: (x, y, z) \to (-x, -y, z), \tag{312a}$$
$$S_2': (x, y, z) \to (-x + 1/2, y + 1/2, -z + 1/2), \tag{312b}$$
$$I: (x, y, z) \to (-x, -y, -z). \tag{312c}$$

The $\mathbb{Z}_2$ cohomology ring is given by

$$\mathbb{Z}_2[A_i, A_{c'}, A_c, B_{\beta 1}, B_{\beta 2}, B_{\beta 3}, C_\gamma]/\langle \mathscr{R}_2, \mathscr{R}_3, \mathscr{R}_4, \mathscr{R}_5, \mathscr{R}_6 \rangle \tag{313}$$

where the relations are

$$\mathscr{R}_2: \quad A_{c'}A_i, \quad A_c A_{c'}, \quad A_{c'}^2, \tag{314a}$$

$$\mathscr{R}_3: \quad A_{c'}B_{\beta 1}, \quad A_{c'}B_{\beta 2}, \quad A_c B_{\beta 1} + A_i B_{\beta 1} + A_c B_{\beta 2}, \quad A_{c'}B_{\beta 3}, \tag{314b}$$

$$\mathscr{R}_4: \quad A_{c'}C_\gamma, \quad B_{\beta 1}(A_c^2 + A_c A_i + B_{\beta 1}), \quad B_{\beta 1}(A_c^2 + A_i^2 + B_{\beta 2}), \quad A_c^2 B_{\beta 3} + A_c A_i B_{\beta 3} + B_{\beta 1}B_{\beta 3} + A_c C_\gamma,$$
$$\quad A_c^2 B_{\beta 1} + A_c A_i B_{\beta 1} + A_i^2 B_{\beta 2} + B_{\beta 2}^2, \quad A_c^2 B_{\beta 3} + A_i^2 B_{\beta 3} + B_{\beta 2}B_{\beta 3} + A_c C_\gamma + A_i C_\gamma, \quad B_{\beta 3}(A_c A_i + A_i^2 + B_{\beta 3}), \tag{314c}$$

$$\mathscr{R}_5: \quad B_{\beta 1}C_\gamma, \quad B_{\beta 2}C_\gamma, \quad (A_c A_i + A_i^2 + B_{\beta 3})C_\gamma, \tag{314d}$$

$$\mathscr{R}_6: \quad C_\gamma(A_c^2 A_i + A_i^3 + C_\gamma). \tag{314e}$$

We have the following table regarding IWPs and group cohomology at degree 3.

| Wyckoff position | Little group Intl. | Little group Schönflies | Coordinates | LSM anomaly class | Topo. inv. |
|---|---|---|---|---|---|
| 2a | $2/m$ | $C_{2h}$ | $(0,0,0),\ (1/2,1/2,1/2)$ | $A_c^2 A_i + A_i^3 + A_i B_{\beta 2} + C_\gamma$ | $\varphi_1[I]$ |
| 2b | $2/m$ | $C_{2h}$ | $(0,0,1/2),\ (1/2,1/2,0)$ | $C_\gamma$ | $\varphi_1[T_3 I]$ |
| 2c | $2/m$ | $C_{2h}$ | $(0,1/2,0),\ (1/2,0,1/2)$ | $A_i B_{\beta 2} + A_c B_{\beta 3} + A_i B_{\beta 3} + C_\gamma$ | $\varphi_1[T_2 I]$ |
| 2d | $2/m$ | $C_{2h}$ | $(0,1/2,1/2),\ (1/2,0,0)$ | $A_c B_{\beta 3} + A_i B_{\beta 3} + C_\gamma$ | $\varphi_1[T_1 I]$ |

**No. 59:** *Pmmn*

This group is generated by three translations $T_{1,2,3}$ as given in Eqs. (88), a two-fold rotation $C_2$, a two-fold screw $S_2'$, and an inversion $I$:

$$C_2\colon (x,y,z) \to (-x+1/2, -y+1/2, z), \tag{315a}$$
$$S_2'\colon (x,y,z) \to (-x, y+1/2, -z), \tag{315b}$$
$$I\colon (x,y,z) \to (-x, -y, -z). \tag{315c}$$

The $\mathbb{Z}_2$ cohomology ring is given by

$$\mathbb{Z}_2[A_i, A_{c'}, A_c, A_z, B_\beta]/\langle \mathcal{R}_2, \mathcal{R}_3, \mathcal{R}_4 \rangle \tag{316}$$

where the relations are

$$\mathcal{R}_2\colon \quad A_c(A_{c'}+A_i), \quad A_{c'}(A_{c'}+A_i), \quad A_z(A_{c'}+A_i+A_z), \tag{317a}$$
$$\mathcal{R}_3\colon \quad (A_{c'}+A_i)B_\beta, \tag{317b}$$
$$\mathcal{R}_4\colon \quad B_\beta(A_c^2 + A_cA_i + B_\beta). \tag{317c}$$

We have the following table regarding IWPs and group cohomology at degree 3.

| Wyckoff position | Little group Intl. | Schönflies | Coordinates | LSM anomaly class | Topo. inv. |
|---|---|---|---|---|---|
| 2a | $mm2$ | $C_{2v}$ | $(1/4,1/4,z)$, $(3/4,3/4,-z)$ | $A_zB_\beta$ | $\varphi_2[T_3, C_2]$ |
| 2b | $mm2$ | $C_{2v}$ | $(1/4,3/4,z)$, $(3/4,1/4,-z)$ | $A_z(A_c^2 + A_cA_i + B_\beta)$ | $\varphi_2[T_3, T_1C_2]$ |
| 4c | $\bar{1}$ | $C_i$ | $(0,0,0)$, $(1/2,1/2,0)$, $(0,1/2,0)$, $(1/2,0,0)$ | $A_i(A_{c'}+A_i)(A_i+A_z)$ | $\varphi_1[I]$ |
| 4d | $\bar{1}$ | $C_i$ | $(0,0,1/2)$, $(1/2,1/2,1/2)$, $(0,1/2,1/2)$, $(1/2,0,1/2)$ | $A_i(A_{c'}+A_i)A_z$ | $\varphi_1[T_3I]$ |

**No. 60:** *Pbcn*

This group is generated by three translations $T_{1,2,3}$ as given in Eqs. (88), a two-fold screw $S_2$, a two-fold rotation $C_2'$, and an inversion $I$:

$$S_2\colon (x,y,z) \to (-x+1/2, -y+1/2, z+1/2), \tag{318a}$$
$$C_2'\colon (x,y,z) \to (-x, y, -z+1/2), \tag{318b}$$
$$I\colon (x,y,z) \to (-x, -y, -z). \tag{318c}$$

The $\mathbb{Z}_2$ cohomology ring is given by

$$\mathbb{Z}_2[A_i, A_{c'}, A_c, B_\beta]/\langle \mathcal{R}_2, \mathcal{R}_3, \mathcal{R}_4 \rangle \tag{319}$$

where the relations are

$$\mathcal{R}_2\colon \quad A_cA_i, \quad A_cA_{c'}, \quad A_c^2 + A_{c'}A_i, \tag{320a}$$
$$\mathcal{R}_3\colon \quad A_cB_\beta, \tag{320b}$$
$$\mathcal{R}_4\colon \quad B_\beta(A_i^2 + B_\beta). \tag{320c}$$

We have the following table regarding IWPs and group cohomology at degree 3.

| Wyckoff position | Little group Intl. | Schönflies | Coordinates | LSM anomaly class | Topo. inv. |
|---|---|---|---|---|---|
| 4a | $\bar{1}$ | $C_i$ | $(0,0,0)$, $(1/2,1/2,1/2)$, $(0,0,1/2)$, $(1/2,1/2,0)$ | $A_iB_\beta$ | $\varphi_1[I]$ |
| 4b | $\bar{1}$ | $C_i$ | $(0,1/2,0)$, $(1/2,0,1/2)$, $(0,1/2,1/2)$, $(1/2,0,0)$ | $A_i(A_i^2 + B_\beta)$ | $\varphi_1[T_1I]$ |
| 4c | $2$ | $C_2$ | $(0,y,1/4)$, $(1/2,-y+1/2,3/4)$, $(0,-y,3/4)$, $(1/2,y+1/2,1/4)$ | $A_{c'}B_\beta$ | $\varphi_2[T_2, C_2']$ |

**No. 61:** *Pbca*

This group is generated by three translations $T_{1,2,3}$ as given in Eqs. (88), a two-fold screw $S_2$, a two-fold screw $S_2'$, and an inversion $I$:

$$S_2 \colon (x,y,z) \to (-x+1/2, -y, z+1/2), \tag{321a}$$
$$S_2' \colon (x,y,z) \to (-x, y+1/2, -z+1/2), \tag{321b}$$
$$I \colon (x,y,z) \to (-x, -y, -z). \tag{321c}$$

The $\mathbb{Z}_2$ cohomology ring is given by

$$\mathbb{Z}_2[A_i, A_{c'}, A_c, C_\beta]/\langle \mathscr{R}_2, \mathscr{R}_4, \mathscr{R}_6 \rangle \tag{322}$$

where the relations are

$$\mathscr{R}_2 \colon \quad A_c(A_{c'} + A_i), \quad A_{c'}^2 + A_c A_i + A_{c'} A_i, \quad A_c^2 + A_{c'} A_i, \tag{323a}$$
$$\mathscr{R}_4 \colon \quad A_{c'} C_\beta, \quad A_c C_\beta, \tag{323b}$$
$$\mathscr{R}_6 \colon \quad C_\beta(A_i^3 + C_\beta). \tag{323c}$$

We have the following table regarding IWPs and group cohomology at degree 3.

| Wyckoff position | Little group Intl. | Schönflies | Coordinates | LSM anomaly class | Topo. inv. |
|---|---|---|---|---|---|
| 4a | $\bar{1}$ | $C_i$ | $(0,0,0)$, $(1/2,0,1/2)$, $(0,1/2,1/2)$, $(1/2,1/2,0)$ | $A_i^3 + C_\beta$ | $\varphi_1[I]$ |
| 4b | $\bar{1}$ | $C_i$ | $(0,0,1/2)$, $(1/2,0,0)$, $(0,1/2,0)$, $(1/2,1/2,1/2)$ | $C_\beta$ | $\varphi_1[T_1 I]$ |

**No. 62:** *Pnma*

This group is generated by three translations $T_{1,2,3}$ as given in Eqs. (88), a two-fold screw $S_2$, a two-fold screw $S_2'$, and an inversion $I$:

$$S_2 \colon (x,y,z) \to (-x+1/2, -y, z+1/2), \tag{324a}$$
$$S_2' \colon (x,y,z) \to (-x, y+1/2, -z), \tag{324b}$$
$$I \colon (x,y,z) \to (-x, -y, -z). \tag{324c}$$

The $\mathbb{Z}_2$ cohomology ring is given by

$$\mathbb{Z}_2[A_i, A_{c'}, A_c, B_{\beta 1}, B_{\beta 2}]/\langle \mathscr{R}_2, \mathscr{R}_3, \mathscr{R}_4 \rangle \tag{325}$$

where the relations are

$$\mathscr{R}_2 \colon \quad A_c(A_{c'} + A_i), \quad A_{c'}^2 + A_c A_i + A_{c'} A_i, \quad A_c^2, \tag{326a}$$
$$\mathscr{R}_3 \colon \quad (A_c + A_{c'} + A_i)B_{\beta 1}, \quad A_i B_{\beta 1} + A_{c'} B_{\beta 2}, \quad A_{c'} B_{\beta 1} + A_i B_{\beta 1} + A_c B_{\beta 2}, \tag{326b}$$
$$\mathscr{R}_4 \colon \quad B_{\beta 1}(A_{c'} A_i + A_i^2 + B_{\beta 1}), \quad B_{\beta 1}(A_{c'} A_i + A_i^2 + B_{\beta 2}), \quad A_i^2 B_{\beta 1} + A_i^2 B_{\beta 2} + B_{\beta 2}^2. \tag{326c}$$

We have the following table regarding IWPs and group cohomology at degree 3.

| Wyckoff position | Little group Intl. | Schönflies | Coordinates | LSM anomaly class | Topo. inv. |
|---|---|---|---|---|---|
| 4a | $\bar{1}$ | $C_i$ | $(0,0,0)$, $(1/2,0,1/2)$, $(0,1/2,0)$, $(1/2,1/2,1/2)$ | $A_{c'}^2 A_i + A_i^3 + A_{c'} B_{\beta 1} + A_i B_{\beta 2}$ | $\varphi_1[I]$ |
| 4b | $\bar{1}$ | $C_i$ | $(0,0,1/2)$, $(1/2,0,0)$, $(0,1/2,1/2)$, $(1/2,1/2,0)$ | $A_{c'} B_{\beta 1} + A_i B_{\beta 2}$ | $\varphi_1[T_1 I]$ |
| 4c | $m$ | $C_s$ | $(x,1/4,z)$, $(-x+1/2, 3/4, z+1/2)$, $(-x, 3/4, -z)$, $(x+1/2, 1/4, -z+1/2)$ | $(A_{c'} + A_i)B_{\beta 1}$ | $\widetilde{\varphi_3}[T_3, S_2 I, S_2' I]$ |

The expression of $\widetilde{\varphi_3}$ is given in Eq. (87).

## No. 63: $Cmcm$

This group is generated by three translations $T_{1,2,3}$ as given in Eqs. (89), a two-fold screw $S_2$, a two-fold rotation $C_2'$, and an inversion $I$:

$$S_2\colon (x,y,z) \to (-x,-y,z+1/2), \tag{327a}$$
$$C_2'\colon (x,y,z) \to (-x,y,-z+1/2), \tag{327b}$$
$$I\colon (x,y,z) \to (-x,-y,-z). \tag{327c}$$

The $\mathbb{Z}_2$ cohomology ring is given by

$$\mathbb{Z}_2[A_i, A_{c'}, A_c, A_{x+y}, B_{xy}]/\langle \mathscr{R}_2, \mathscr{R}_3, \mathscr{R}_4 \rangle \tag{328}$$

where the relations are

$$\mathscr{R}_2\colon \quad A_{c'}A_{x+y}, \quad (A_c + A_{c'})(A_c + A_i), \quad A_{x+y}(A_c + A_i + A_{x+y}), \tag{329a}$$
$$\mathscr{R}_3\colon \quad A_{x+y}B_{xy}, \tag{329b}$$
$$\mathscr{R}_4\colon \quad B_{xy}(A_cA_i + A_i^2 + B_{xy}). \tag{329c}$$

We have the following table regarding IWPs and group cohomology at degree 3.

| Wyckoff position | Little group Intl. | Schönflies | Coordinates $(0,0,0) + (1/2,1/2,0)+$ | LSM anomaly class | Topo. inv. |
|---|---|---|---|---|---|
| 4a | $2/m$ | $C_{2h}$ | $(0,0,0), (0,0,1/2)$ | $(A_c + A_i)(A_{c'}A_i + A_i^2 + A_iA_{x+y} + B_{xy})$ | $\varphi_1[I]$ |
| 4b | $2/m$ | $C_{2h}$ | $(0,1/2,0), (0,1/2,1/2)$ | $(A_c + A_i)B_{xy}$ | $\varphi_1[T_1T_2I]$ |
| 4c | $mm2$ | $C_{2v}$ | $(0,y,1/4), (0,-y,3/4)$ | $(A_c + A_{c'})B_{xy}$ | $\varphi_2[T_1T_2, C_2']$ |
| 8d | $\bar{1}$ | $C_i$ | $(1/4,1/4,0), (3/4,3/4,1/2),$ $(3/4,1/4,1/2), (1/4,3/4,0)$ | $A_i(A_c + A_i)A_{x+y}$ | $\varphi_1[T_1I]$ |

## No. 64: $Cmce$

This group is generated by three translations $T_{1,2,3}$ as given in Eqs. (89), a two-fold screw $S_2$, a two-fold rotation $C_2'$, and an inversion $I$:

$$S_2\colon (x,y,z) \to (-x,-y+1/2,z+1/2), \tag{330a}$$
$$C_2'\colon (x,y,z) \to (-x+1/2,y,-z+1/2), \tag{330b}$$
$$I\colon (x,y,z) \to (-x,-y,-z). \tag{330c}$$

The $\mathbb{Z}_2$ cohomology ring is given by

$$\mathbb{Z}_2[A_i, A_{c'}, A_c, A_{x+y}, C_\gamma]/\langle \mathscr{R}_2, \mathscr{R}_4, \mathscr{R}_6 \rangle \tag{331}$$

where the relations are

$$\mathscr{R}_2\colon \quad A_cA_{c'} + A_cA_i + A_{c'}A_i + A_{c'}A_{x+y}, \quad (A_c + A_{c'})(A_c + A_i), \quad (A_c + A_{x+y})(A_i + A_{x+y}), \tag{332a}$$
$$\mathscr{R}_4\colon \quad (A_c + A_{c'})C_\gamma, \quad (A_{c'} + A_{x+y})C_\gamma, \tag{332b}$$
$$\mathscr{R}_6\colon \quad C_\gamma(A_{c'}^2A_i + A_i^3 + C_\gamma). \tag{332c}$$

We have the following table regarding IWPs and group cohomology at degree 3.

| Wyckoff position | Little group Intl. | Schönflies | Coordinates $(0,0,0) + (1/2,1/2,0)+$ | LSM anomaly class | Topo. inv. |
|---|---|---|---|---|---|
| 4a | $2/m$ | $C_{2h}$ | $(0,0,0), (0,1/2,1/2)$ | $A_i(A_{c'} + A_i)(A_i + A_{x+y}) + C_\gamma$ | $\varphi_1[I]$ |
| 4b | $2/m$ | $C_{2h}$ | $(1/2,0,0), (1/2,1/2,1/2)$ | $C_\gamma$ | $\varphi_1[T_1T_2I]$ |
| 8c | $\bar{1}$ | $C_i$ | $(1/4,1/4,0), (3/4,1/4,1/2),$ $(3/4,3/4,1/2), (1/4,3/4,0)$ | $A_i^2(A_{c'} + A_{x+y})$ | $\varphi_1[T_1I]$ |
| 8e | $2$ | $C_2$ | $(1/4,y,1/4), (3/4,-y+1/2,3/4),$ $(3/4,-y,3/4), (1/4,y+1/2,1/4)$ | $A_{c'}A_i(A_{c'} + A_{x+y})$ | $\varphi_2[S_2I, C_2']$ |

**No. 65:** $Cmmm$

This group is generated by three translations $T_{1,2,3}$ as given in Eqs. (89), a two-fold rotation $C_2$, a two-fold rotation $C_2'$, and an inversion $I$:

$$C_2: (x, y, z) \to (-x, -y, z), \tag{333a}$$
$$C_2': (x, y, z) \to (-x, y, -z), \tag{333b}$$
$$I: (x, y, z) \to (-x, -y, -z). \tag{333c}$$

The $\mathbb{Z}_2$ cohomology ring is given by

$$\mathbb{Z}_2[A_i, A_{c'}, A_c, A_{x+y}, A_z, B_{xy}]/\langle \mathcal{R}_2, \mathcal{R}_3, \mathcal{R}_4 \rangle \tag{334}$$

where the relations are

$$\mathcal{R}_2: \quad A_{c'}A_{x+y}, \quad A_{x+y}(A_c + A_i + A_{x+y}), \quad A_z(A_{c'} + A_i + A_z), \tag{335a}$$
$$\mathcal{R}_3: \quad A_{x+y}B_{xy}, \tag{335b}$$
$$\mathcal{R}_4: \quad B_{xy}(A_c^2 + A_cA_{c'} + A_{c'}A_i + A_i^2 + B_{xy}). \tag{335c}$$

We have the following table regarding IWPs and group cohomology at degree 3.

| Wyckoff position | Little group Intl. | Schönflies | Coordinates $(0,0,0) + (1/2,1/2,0)+$ | LSM anomaly class | Topo. inv. |
|---|---|---|---|---|---|
| 2a | $mmm$ | $D_{2h}$ | $(0,0,0)$ | $(A_{c'} + A_i + A_z)((A_c + A_i)(A_c + A_{c'} + A_i + A_{x+y}) + B_{xy})$ | $\varphi_2[C_2, C_2']$ |
| 2b | $mmm$ | $D_{2h}$ | $(1/2,0,0)$ | $(A_{c'} + A_i + A_z)B_{xy}$ | $\varphi_2[T_1T_2C_2, C_2']$ |
| 2c | $mmm$ | $D_{2h}$ | $(1/2,0,1/2)$ | $A_zB_{xy}$ | $\varphi_2[T_1T_2C_2, T_3C_2']$ |
| 2d | $mmm$ | $D_{2h}$ | $(0,0,1/2)$ | $A_z((A_c + A_i)(A_c + A_{c'} + A_i + A_{x+y}) + B_{xy})$ | $\varphi_2[C_2, T_3C_2']$ |
| 4e | $2/m$ | $C_{2h}$ | $(1/4,1/4,0), (3/4,1/4,0)$ | $(A_c + A_i)A_{x+y}(A_i + A_z)$ | $\varphi_1[T_1I]$ |
| 4f | $2/m$ | $C_{2h}$ | $(1/4,1/4,1/2), (3/4,1/4,1/2)$ | $(A_c + A_i)A_{x+y}A_z$ | $\varphi_1[T_1T_3I]$ |

**No. 66:** $Cccm$

This group is generated by three translations $T_{1,2,3}$ as given in Eqs. (89), a two-fold rotation $C_2$, a two-fold rotation $C_2'$, and an inversion $I$:

$$C_2: (x, y, z) \to (-x, -y, z), \tag{336a}$$
$$C_2': (x, y, z) \to (-x, y, -z + 1/2), \tag{336b}$$
$$I: (x, y, z) \to (-x, -y, -z). \tag{336c}$$

The $\mathbb{Z}_2$ cohomology ring is given by

$$\mathbb{Z}_2[A_i, A_{c'}, A_c, A_{x+y}, B_{xy}, B_{z(x+y)}]/\langle \mathcal{R}_2, \mathcal{R}_3, \mathcal{R}_4 \rangle \tag{337}$$

where the relations are

$$\mathcal{R}_2: \quad A_{c'}A_i, \quad A_{c'}A_{x+y}, \quad A_{x+y}(A_c + A_i + A_{x+y}), \tag{338a}$$
$$\mathcal{R}_3: \quad A_{x+y}B_{xy}, \quad A_{c'}B_{z(x+y)}, \quad A_iB_{xy} + A_cB_{z(x+y)} + A_iB_{z(x+y)} + A_{x+y}B_{z(x+y)}, \tag{338b}$$
$$\mathcal{R}_4: \quad B_{xy}(A_c^2 + A_cA_{c'} + A_i^2 + B_{xy}), \quad B_{xy}(A_cA_i + A_i^2 + B_{z(x+y)}), \quad B_{z(x+y)}(A_cA_i + A_i^2 + B_{z(x+y)}). \tag{338c}$$

We have the following table regarding IWPs and group cohomology at degree 3.

| Wyckoff position | Little group Intl. | Schönflies | Coordinates $(0,0,0) + (1/2,1/2,0)+$ | LSM anomaly class | Topo. inv. |
|---|---|---|---|---|---|
| 4a | $222$ | $D_2$ | $(0,0,1/4), (0,0,3/4)$ | $A_{c'}(A_c^2 + A_cA_{c'} + B_{xy})$ | $\varphi_2[C_2, C_2']$ |
| 4b | $222$ | $D_2$ | $(0,1/2,1/4), (0,1/2,3/4)$ | $A_{c'}B_{xy}$ | $\varphi_2[T_1T_2C_2, C_2']$ |
| 4c | $2/m$ | $C_{2h}$ | $(0,0,0), (0,0,1/2)$ | $A_i(A_c^2 + A_i^2 + A_cA_{x+y} + A_iA_{x+y} + B_{xy})$ | $\varphi_1[I]$ |
| 4d | $2/m$ | $C_{2h}$ | $(0,1/2,0), (0,1/2,1/2)$ | $A_iB_{xy}$ | $\varphi_1[T_1T_2I]$ |
| 4e | $2/m$ | $C_{2h}$ | $(1/4,1/4,0), (3/4,1/4,1/2)$ | $A_iB_{xy} + A_cB_{z(x+y)} + A_iB_{z(x+y)}$ | $\varphi_1[T_1I]$ |
| 4f | $2/m$ | $C_{2h}$ | $(1/4,3/4,0), (3/4,3/4,1/2)$ | $A_i(A_cA_{x+y} + A_iA_{x+y} + B_{xy}) + (A_c + A_i)B_{z(x+y)}$ | $\varphi_1[T_2I]$ |

**No. 67:** *Cmme*

This group is generated by three translations $T_{1,2,3}$ as given in Eqs. (89), a two-fold rotation $C_2$, a two-fold rotation $C_2'$, and an inversion $I$:

$$C_2 : (x, y, z) \to (-x + 1/2, -y, z), \tag{339a}$$
$$C_2' : (x, y, z) \to (-x + 1/2, y, -z), \tag{339b}$$
$$I : (x, y, z) \to (-x, -y, -z). \tag{339c}$$

The $\mathbb{Z}_2$ cohomology ring is given by

$$\mathbb{Z}_2[A_i, A_{c'}, A_c, A_{x+y}, A_z]/\langle \mathscr{R}_2 \rangle \tag{340}$$

where the relations are

$$\mathscr{R}_2 : \quad A_c A_i + A_{c'} A_i + A_{c'} A_{x+y}, \quad A_{x+y}(A_c + A_i + A_{x+y}), \quad A_z(A_{c'} + A_i + A_z). \tag{341a}$$

We have the following table regarding IWPs and group cohomology at degree 3.

| Wyckoff position | Little group Intl. | Schönflies | Coordinates $(0,0,0) + (1/2,1/2,0)+$ | LSM anomaly class | Topo. inv. |
|---|---|---|---|---|---|
| 4a | 222 | $D_2$ | $(1/4,0,0)$, $(3/4,0,0)$ | $A_c(A_c A_{c'} + A_{c'}^2 + A_c A_i + A_{c'} A_i + A_c A_z + A_{c'} A_z + A_{x+y} A_z)$ | $\varphi_2[C_2, C_2']$ |
| 4b | 222 | $D_2$ | $(1/4,0,1/2)$, $(3/4,0,1/2)$ | $A_c(A_c + A_{c'} + A_{x+y})A_z$ | $\varphi_2[C_2, T_3 C_2']$ |
| 4c | $2/m$ | $C_{2h}$ | $(0,0,0)$, $(0,1/2,0)$ | $A_i(A_c A_{c'} + A_i^2 + A_i A_{x+y} + A_c A_z + A_i A_z + A_{x+y} A_z)$ | $\varphi_1[I]$ |
| 4d | $2/m$ | $C_{2h}$ | $(0,0,1/2)$, $(0,1/2,1/2)$ | $A_i(A_c + A_i + A_{x+y})A_z$ | $\varphi_1[T_3 I]$ |
| 4e | $2m$ | $C_{2h}$ | $(1/4,1/4,0)$, $(3/4,1/4,0)$ | $A_i(A_c A_{c'} + A_{c'}^2 + A_i A_{x+y} + A_c A_z + A_{c'} A_z + A_{x+y} A_z)$ | $\varphi_1[T_1 I]$ |
| 4f | $2m$ | $C_{2h}$ | $(1/4,1/4,1/2)$, $(3/4,1/4,1/2)$ | $A_i(A_c + A_{c'} + A_{x+y})A_z$ | $\varphi_1[T_1 T_3 I]$ |
| 4g | $mm2$ | $C_{2v}$ | $(0,1/4,z)$, $(0,3/4,-z)$ | $A_c A_{x+y} A_z$ | $\varphi_2[T_3, T_2 C_2]$ |

**No. 68:** *Ccce*

This group is generated by three translations $T_{1,2,3}$ as given in Eqs. (89), a two-fold rotation $C_2$, a two-fold rotation $C_2'$, and an inversion $I$:

$$C_2 : (x, y, z) \to (-x, -y + 1/2, z), \tag{342a}$$
$$C_2' : (x, y, z) \to (-x, y, -z + 1/2), \tag{342b}$$
$$I : (x, y, z) \to (-x, -y, -z). \tag{342c}$$

The $\mathbb{Z}_2$ cohomology ring is given by

$$\mathbb{Z}_2[A_i, A_{c'}, A_c, A_{x+y}, C_\gamma]/\langle \mathscr{R}_2, \mathscr{R}_4, \mathscr{R}_6 \rangle \tag{343}$$

where the relations are

$$\mathscr{R}_2 : \quad A_{c'} A_i, \quad A_c A_i + A_{c'} A_{x+y}, \quad (A_c + A_{x+y})(A_i + A_{x+y}), \tag{344a}$$
$$\mathscr{R}_4 : \quad A_i C_\gamma, \quad A_{x+y} C_\gamma, \tag{344b}$$
$$\mathscr{R}_6 : \quad C_\gamma(A_c^2 A_{c'} + A_c A_{c'}^2 + C_\gamma). \tag{344c}$$

We have the following table regarding IWPs and group cohomology at degree 3.

| Wyckoff position | Little group Intl. | Schönflies | Coordinates $(0,0,0) + (1/2,1/2,0)+$ | LSM anomaly class | Topo. inv. |
|---|---|---|---|---|---|
| 4a | 222 | $D_2$ | $(0,1/4,1/4)$, $(0,3/4,3/4)$ | $A_c^2 A_{c'} + A_c A_{c'}^2 + A_c^2 A_i + C_\gamma$ | $\varphi_2[C_2, C_2']$ |
| 4b | 222 | $D_2$ | $(0,1/4,3/4)$, $(0,3/4,1/4)$ | $C_\gamma$ | $\varphi_2[T_1 T_2 C_2, C_2']$ |
| 8c | $\bar{1}$ | $C_i$ | $(1/4,3/4,0)$, $(1/4,1/4,0)$, $(3/4,3/4,1/2)$, $(3/4,1/4,1/2)$ | $A_i^2 A_{x+y}$ | $\varphi_1[T_1 I]$ |
| 8d | $\bar{1}$ | $C_i$ | $(0,0,0)$, $(1/2,0,0)$, $(0,0,1/2)$, $(1/2,0,1/2)$ | $A_i^2(A_i + A_{x+y})$ | $\varphi_1[I]$ |
| 8h | $2$ | $C_2$ | $(1/4,0,z)$, $(3/4,0,-z+1/2)$, $(3/4,0,-z)$, $(1/4,0,z+1/2)$ | $A_c^2 A_i$ | $\varphi_2[T_2^{-1} C_2 C_2' I, T_2^{-1} C_2]$ |

**No. 69:** *Fmmm*

This group is generated by three translations $T_{1,2,3}$ as given in Eqs. (92), a two-fold rotation $C_2$, a two-fold rotation $C_2'$, and an inversion $I$:

$$C_2\colon (x,y,z) \to (-x,-y,z), \tag{345a}$$
$$C_2'\colon (x,y,z) \to (-x,y,-z), \tag{345b}$$
$$I\colon (x,y,z) \to (-x,-y,-z). \tag{345c}$$

The $\mathbb{Z}_2$ cohomology ring is given by

$$\mathbb{Z}_2[A_i, A_{c'}, A_c, A_{x+z}, A_{y+z}, C_{xyz}]/\langle \mathcal{R}_2, \mathcal{R}_4, \mathcal{R}_6\rangle \tag{346}$$

where the relations are

$$\mathcal{R}_2\colon \quad A_c A_{x+z} + A_{c'} A_{x+z} + A_c A_{y+z}, \quad A_{x+z}(A_c + A_i + A_{x+z}), \quad A_c A_{x+z} + A_{c'} A_{x+z} + A_{c'} A_{y+z} + A_i A_{y+z} + A_{y+z}^2, \tag{347a}$$

$$\mathcal{R}_4\colon \quad A_{x+z} C_{xyz}, \quad A_{y+z} C_{xyz}, \tag{347b}$$

$$\mathcal{R}_6\colon \quad C_{xyz}(A_c^2 A_{c'} + A_c A_{c'}^2 + A_c^2 A_i + A_c A_{c'} A_i + A_{c'}^2 A_i + A_i^3 + C_{xyz}). \tag{347c}$$

We have the following table regarding IWPs and group cohomology at degree 3.

| Wyckoff position | Little group Intl. | Little group Schönflies | Coordinates $(0,0,0)+$ $(0,1/2,1/2)+$ $(1/2,0,1/2)+$ $(1/2,1/2,0)+$ | LSM anomaly class | Topo. inv. |
|---|---|---|---|---|---|
| 4a | $mmm$ | $D_{2h}$ | $(0,0,0)$ | $A_c^2 A_{c'} + A_c A_{c'}^2 + A_c^2 A_i + A_c A_{c'} A_i + A_{c'}^2 A_i + A_i^3 + A_{c'} A_i A_{x+z} + A_i^2 A_{x+z}$ $+ A_c A_{c'} A_{y+z} + A_{c'} A_i A_{y+z} + A_i^2 A_{y+z} + A_i A_{x+z} A_{y+z} + C_{xyz}$ | $\varphi_2[C_2, C_2']$ |
| 4b | $mmm$ | $D_{2h}$ | $(0,0,1/2)$ | $C_{xyz}$ | $\varphi_2[C_2, T_1 T_2 T_3^{-1} C_2']$ |
| 8c | $2/m$ | $C_{2h}$ | $(0,1/4,1/4),\ (0,3/4,1/4)$ | $A_i A_{x+z}(A_c + A_i + A_{y+z})$ | $\varphi_1[T_1 I]$ |
| 8d | $2m$ | $C_{2h}$ | $(1/4,0,1/4),\ (3/4,0,1/4)$ | $A_i(A_c A_{x+z} + A_{c'} A_{x+z} + A_{c'} A_{y+z} + A_i A_{y+z} + A_{x+z} A_{y+z})$ | $\varphi_1[T_2 I]$ |
| 8e | $2/m$ | $C_{2h}$ | $(1/4,1/4,0),\ (3/4,1/4,0)$ | $A_i A_{x+z} A_{y+z}$ | $\varphi_1[T_3 I]$ |
| 8f | $222$ | $D_2$ | $(1/4,1/4,1/4),\ (3/4,3/4,3/4)$ | $A_c A_{c'} A_{y+z}$ | $\varphi_2[T_3 C_2, T_2 C_2']$ |

**No. 70:** *Fddd*

This group is generated by three translations $T_{1,2,3}$ as given in Eqs. (92), a two-fold rotation $C_2$, a two-fold rotation $C_2'$, and an inversion $I$:

$$C_2\colon (x,y,z) \to (-x+1/4,-y+1/4,z), \tag{348a}$$
$$C_2'\colon (x,y,z) \to (-x+1/4,y,-z+1/4), \tag{348b}$$
$$I\colon (x,y,z) \to (-x,-y,-z). \tag{348c}$$

The $\mathbb{Z}_2$ cohomology ring is given by

$$\mathbb{Z}_2[A_i, A_{c'}, A_c, B_{xy+xz+yz}, C_\gamma]/\langle \mathcal{R}_2, \mathcal{R}_3, \mathcal{R}_4, \mathcal{R}_5, \mathcal{R}_6\rangle \tag{349}$$

where the relations are

$$\mathcal{R}_2\colon \quad A_{c'} A_i, \quad A_c A_i, \tag{350a}$$

$$\mathcal{R}_3\colon \quad A_{c'}(A_c A_{c'} + B_{xy+xz+yz}), \quad A_c(A_c A_{c'} + B_{xy+xz+yz}), \tag{350b}$$

$$\mathcal{R}_4\colon \quad A_i C_\gamma, \quad A_c^2 A_{c'}^2 + A_i^2 B_{xy+xz+yz} + B_{xy+xz+yz}^2, \tag{350c}$$

$$\mathcal{R}_5\colon \quad (A_c A_{c'} + B_{xy+xz+yz}) C_\gamma, \tag{350d}$$

$$\mathcal{R}_6\colon \quad C_\gamma(A_c^2 A_{c'} + A_c A_{c'}^2 + C_\gamma). \tag{350e}$$

We have the following table regarding IWPs and group cohomology at degree 3.

| Wyckoff position | Little group Intl. | Schönflies | Coordinates $(0,0,0)+$ $(0,1/2,1/2)+$ $(1/2,0,1/2)+$ $(1/2,1/2,0)+$ | LSM anomaly class | Topo. inv. |
|---|---|---|---|---|---|
| 8a | 222 | $D_2$ | $(1/8,1/8,1/8)$, $(7/8,7/8,7/8)$ | $A_c^2 A_{c'} + A_c A_{c'}^2 + C_\gamma$ | $\varphi_2[C_2, C_2']$ |
| 8b | 222 | $D_2$ | $(1/8,1/8,5/8)$, $(7/8,7/8,3/8)$ | $C_\gamma$ | $\varphi_2[C_2, T_1 T_2 T_3^{-1} C_2']$ |
| 16c | $\bar{1}$ | $C_i$ | $(0,0,0)$, $(3/4,3/4,0)$, $(3/4,0,3/4)$, $(0,3/4,3/4)$ | $A_i(A_i^2 + B_{xy+xz+yz})$ | $\varphi_1[I]$ |
| 16d | $\bar{1}$ | $C_i$ | $(1/2,1/2,1/2)$, $(1/4,1/4,1/2)$, $(1/4,1/2,1/4)$, $(1/2,1/4,1/4)$ | $A_i B_{xy+xz+yz}$ | $\varphi_1[T_1 T_2 T_3 I]$ |

## No. 71: *Immm*

This group is generated by three translations $T_{1,2,3}$ as given in Eqs. (91), a two-fold rotation $C_2$, a two-fold rotation $C_2'$, and an inversion $I$:

$$C_2 \colon (x,y,z) \to (-x,-y,z), \tag{351a}$$
$$C_2' \colon (x,y,z) \to (-x,y,-z), \tag{351b}$$
$$I \colon (x,y,z) \to (-x,-y,-z). \tag{351c}$$

The $\mathbb{Z}_2$ cohomology ring is given by

$$\mathbb{Z}_2[A_i, A_{c'}, A_c, A_{x+y+z}, B_\beta, B_{x(y+z)}, B_{y(x+z)}, C_{xyz}]/\langle \mathscr{R}_2, \mathscr{R}_3, \mathscr{R}_4, \mathscr{R}_5, \mathscr{R}_6 \rangle \tag{352}$$

where the relations are

$$\mathscr{R}_2 \colon \quad A_{c'} A_{x+y+z}, \quad A_c A_{x+y+z}, \quad A_{x+y+z}(A_i + A_{x+y+z}), \tag{353a}$$

$$\mathscr{R}_3 \colon \quad A_{x+y+z} B_\beta, \quad A_{x+y+z} B_{x(y+z)}, \quad A_c B_\beta + A_i B_\beta + A_c B_{x(y+z)} + A_c B_{y(x+z)} + A_{c'} B_{y(x+z)}, \quad A_{x+y+z} B_{y(x+z)}, \tag{353b}$$

$$\mathscr{R}_4 \colon \quad A_{x+y+z} C_{xyz}, \quad A_c^2 B_\beta + A_{c'}^2 B_\beta + A_c A_i B_\beta + A_{c'} A_i B_\beta + B_\beta^2 + A_c A_{c'} B_{x(y+z)},$$

$$A_c A_{c'} B_\beta + A_c A_i B_\beta + A_{c'} A_i B_\beta + A_i^2 B_\beta + B_\beta B_{x(y+z)} + A_c C_{xyz} + A_{c'} C_{xyz},$$

$$A_c^2 B_\beta + A_c A_i B_\beta + A_c A_{c'} B_{x(y+z)} + A_{c'}^2 B_{y(x+z)} + A_{c'} A_i B_{y(x+z)} + B_\beta B_{y(x+z)} + A_c C_{xyz},$$

$$B_{x(y+z)}(A_c A_{c'} + A_c A_i + A_{c'} A_i + A_i^2 + B_{x(y+z)}),$$

$$(A_c A_{c'} + A_c A_i + A_{c'} A_i + A_i^2) B_\beta + A_c (A_{c'} + A_i) B_{x(y+z)} + (A_{c'}^2 + A_i^2) B_{y(x+z)} + B_{x(y+z)} B_{y(x+z)} + (A_c + A_i) C_{xyz},$$

$$A_c^2 B_\beta + A_c A_i B_\beta + A_c A_i B_{x(y+z)} + A_{c'} A_i B_{y(x+z)} + A_i^2 B_{y(x+z)} + B_{y(x+z)}^2, \tag{353c}$$

$$\mathscr{R}_5 \colon \quad (A_c^2 + A_{c'}^2 + A_c A_i + A_{c'} A_i + B_\beta) C_{xyz}, \quad B_{x(y+z)} C_{xyz}, \quad (A_c^2 + A_c A_{c'} + A_{c'} A_i + A_i^2 + B_{y(x+z)}) C_{xyz}, \tag{353d}$$

$$\mathscr{R}_6 \colon \quad C_{xyz}(A_c^2 A_{c'} + A_c A_{c'}^2 + A_c^2 A_i + A_c A_{c'} A_i + A_{c'}^2 A_i + A_i^3 + C_{xyz}). \tag{353e}$$

We have the following table regarding IWPs and group cohomology at degree 3.

| Wyckoff position | Little group Intl. | Schönflies | Coordinates $(0,0,0)+$ $(1/2,1/2,1/2)+$ | LSM anomaly class | Topo. inv. |
|---|---|---|---|---|---|
| 2a | $mmm$ | $D_{2h}$ | $(0,0,0)$ | $A_c^2 A_{c'} + A_c A_{c'}^2 + A_c^2 A_i + A_c A_{c'} A_i + A_{c'}^2 A_i + A_i^3$ $+ A_i^2 A_{x+y+z} + (A_c + A_{c'} + A_i) B_{x(y+z)} + C_{xyz}$ | $\varphi_2[C_2, C_2']$ |
| 2b | $mmm$ | $D_{2h}$ | $(0,1/2,1/2)$ | $C_{xyz}$ | $\varphi_2[T_2 T_3 C_2, T_2 T_3 C_2']$ |
| 2c | $mmm$ | $D_{2h}$ | $(1/2,1/2,0)$ | $A_c B_{x(y+z)} + A_{c'} B_{y(x+z)} + A_i B_{y(x+z)} + C_{xyz}$ | $\varphi_2[C_2, T_1 T_2 C_2']$ |
| 2d | $mmm$ | $D_{2h}$ | $(1/2,0,1/2)$ | $(A_{c'} + A_i)(B_{x(y+z)} + B_{y(x+z)}) + C_{xyz}$ | $\varphi_2[T_1 T_3 C_2, C_2']$ |
| 8k | $\bar{1}$ | $C_i$ | $(1/4,1/4,1/4)$, $(3/4,3/4,1/4)$, $(3/4,1/4,3/4)$, $(1/4,3/4,3/4)$ | $A_i^2 A_{x+y+z}$ | $\varphi_1[T_1 T_2 T_3 I]$ |

## No. 72: *Ibam*

This group is generated by three translations $T_{1,2,3}$ as given in Eqs. (91), a two-fold rotation $C_2$, a two-fold rotation $C_2'$, and an inversion $I$:

$$C_2 \colon (x,y,z) \to (-x,-y,z), \tag{354a}$$
$$C_2' \colon (x,y,z) \to (-x,y,-z+1/2), \tag{354b}$$
$$I \colon (x,y,z) \to (-x,-y,-z). \tag{354c}$$

The $\mathbb{Z}_2$ cohomology ring is given by

$$\mathbb{Z}_2[A_i, A_{c'}, A_c, A_{x+y+z}, B_{z(x+y)}]/\langle \mathscr{R}_2, \mathscr{R}_3, \mathscr{R}_4 \rangle \tag{355}$$

where the relations are

$$\mathscr{R}_2: \quad A_{c'}A_{x+y+z}, \quad A_{c'}A_i + A_c A_{x+y+z}, \quad A_{c'}A_i + A_i A_{x+y+z} + A_{x+y+z}^2, \tag{356a}$$

$$\mathscr{R}_3: \quad A_{x+y+z}B_{z(x+y)}, \tag{356b}$$

$$\mathscr{R}_4: \quad B_{z(x+y)}(A_c^2 + A_c A_{c'} + A_i^2 + B_{z(x+y)}). \tag{356c}$$

We have the following table regarding IWPs and group cohomology at degree 3.

| Wyckoff position | Little group Intl. | Little group Schönflies | Coordinates $(0,0,0)+$ $(1/2,1/2,1/2)+$ | LSM anomaly class | Topo. inv. |
|---|---|---|---|---|---|
| 4a | 222 | $D_2$ | $(0,0,1/4)$, $(0,0,3/4)$ | $A_{c'}(A_c^2 + A_c A_{c'} + A_i^2 + B_{z(x+y)})$ | $\varphi_2[C_2, C_2']$ |
| 4b | 222 | $D_2$ | $(1/2,0,1/4)$, $(1/2,0,3/4)$ | $A_{c'}B_{z(x+y)}$ | $\varphi_2[T_2T_3C_2, T_2T_3C_2']$ |
| 4c | $2/m$ | $C_{2h}$ | $(0,0,0)$, $(1/2,1/2,0)$ | $A_i(A_c^2 + A_{c'}A_i + A_i^2 + A_i A_{x+y+z} + B_{z(x+y)})$ | $\varphi_1[I]$ |
| 4d | $2/m$ | $C_{2h}$ | $(1/2,0,0)$, $(0,1/2,0)$ | $A_i B_{z(x+y)}$ | $\varphi_1[T_2T_3I]$ |
| 8e | $\bar{1}$ | $C_i$ | $(1/4,1/4,1/4)$, $(3/4,3/4,1/4)$, $(1/4,3/4,3/4)$, $(3/4,1/4,3/4)$ | $A_i^2(A_{c'} + A_{x+y+z})$ | $\varphi_1[T_1T_2T_3I]$ |

**No. 73:** *Ibca*

This group is generated by three translations $T_{1,2,3}$ as given in Eqs. (91), a two-fold rotation $C_2$, a two-fold rotation $C_2'$, and an inversion $I$:

$$C_2: (x,y,z) \rightarrow (-x, -y+1/2, z), \tag{357a}$$

$$C_2': (x,y,z) \rightarrow (-x+1/2, y, -z), \tag{357b}$$

$$I: (x,y,z) \rightarrow (-x, -y, -z). \tag{357c}$$

The $\mathbb{Z}_2$ cohomology ring is given by

$$\mathbb{Z}_2[A_i, A_{c'}, A_c, A_{x+y+z}]/\langle \mathscr{R}_2 \rangle \tag{358}$$

where the relations are

$$\mathscr{R}_2: \quad A_c A_{c'} + A_c A_i + A_{c'}A_i + A_{c'}A_{x+y+z}, \quad A_c A_{c'} + A_{c'}A_i + A_c A_{x+y+z}, \quad A_c A_{c'} + A_c A_i + A_{c'}A_i + A_i A_{x+y+z} + A_{x+y+z}^2. \tag{359a}$$

We have the following table regarding IWPs and group cohomology at degree 3.

| Wyckoff position | Little group Intl. | Little group Schönflies | Coordinates $(0,0,0)+$ $(1/2,1/2,1/2)+$ | LSM anomaly class | Topo. inv. |
|---|---|---|---|---|---|
| 8a | $\bar{1}$ | $C_i$ | $(0,0,0)$, $(1/2,0,1/2)$, $(0,1/2,1/2)$, $(1/2,1/2,0)$ | $A_i^2(A_i + A_{x+y+z})$ | $\varphi_1[I]$ |
| 8b | $\bar{1}$ | $C_i$ | $(1/4,1/4,1/4)$, $(1/4,3/4,3/4)$, $(3/4,3/4,1/4)$, $(3/4,1/4,3/4)$ | $A_i^2 A_{x+y+z}$ | $\varphi_1[T_1T_2T_3I]$ |
| 8c | 2 | $C_2$ | $(x,0,1/4)$, $(-x+1/2,0,3/4)$, $(-x,0,3/4)$, $(x+1/2,0,1/4)$ | $A_c A_{c'}A_i$ | $\varphi_2[C_2'I, T_2C_2C_2']$ |
| 8d | 2 | $C_2$ | $(1/4,y,0)$, $(1/4,-y,1/2)$, $(3/4,-y,0)$, $(3/4,y,1/2)$ | $A_{c'}(A_c + A_{c'})A_i$ | $\varphi_2[C_2I, C_2']$ |
| 8e | 2 | $C_2$ | $(0,1/4,z)$, $(0,3/4,-z+1/2)$, $(0,3/4,-z)$, $(0,1/4,z+1/2)$ | $A_c(A_c + A_{c'})A_i$ | $\varphi_2[T_2C_2C_2'I, C_2]$ |

**No. 74:** *Imma*

This group is generated by three translations $T_{1,2,3}$ as given in Eqs. (91), a two-fold rotation $C_2$, a two-fold rotation $C_2'$, and an inversion $I$:

$$C_2 : (x, y, z) \rightarrow (-x, -y + 1/2, z), \tag{360a}$$
$$C_2' : (x, y, z) \rightarrow (-x + 1/2, y, -z + 1/2), \tag{360b}$$
$$I : (x, y, z) \rightarrow (-x, -y, -z). \tag{360c}$$

The $\mathbb{Z}_2$ cohomology ring is given by

$$\mathbb{Z}_2[A_i, A_{c'}, A_c, A_{x+y+z}, B_{y(x+z)}, B_{z(x+y)}]/\langle \mathcal{R}_2, \mathcal{R}_3, \mathcal{R}_4 \rangle \tag{361}$$

where the relations are

$$\mathcal{R}_2 : \quad A_c A_{c'} + A_c A_i + A_{c'} A_i + A_{c'} A_{x+y+z}, \quad A_c(A_{c'} + A_{x+y+z}), \quad A_c A_{c'} + A_c A_i + A_i A_{x+y+z} + A_{x+y+z}^2, \tag{362a}$$

$$\mathcal{R}_3 : \quad (A_{c'} + A_{x+y+z})B_{y(x+z)}, \quad A_{c'} B_{y(x+z)} + A_c B_{z(x+y)}, \quad A_{c'} B_{y(x+z)} + A_i B_{y(x+z)} + A_i B_{z(x+y)} + A_{x+y+z} B_{z(x+y)}, \tag{362b}$$

$$\mathcal{R}_4 : \quad A_c A_{c'}^3 + A_c A_{c'}^2 A_i + A_{c'} A_i^3 + A_{c'} A_i^2 A_{x+y+z} + A_{c'} A_i B_{y(x+z)} + A_i^2 B_{y(x+z)} + B_{y(x+z)}^2,$$
$$A_c A_{c'}^3 + A_c A_{c'}^2 A_i + A_{c'} A_i^3 + A_{c'} A_i^2 A_{x+y+z} + A_{c'} A_i B_{y(x+z)} + A_i^2 B_{y(x+z)} + B_{y(x+z)} B_{z(x+y)},$$
$$A_c A_{c'}^3 + A_c A_{c'}^2 A_i + A_{c'} A_i^3 + A_{c'} A_i^2 A_{x+y+z} + A_{c'} A_i B_{z(x+y)} + A_i^2 B_{z(x+y)} + B_{z(x+y)}^2. \tag{362c}$$

We have the following table regarding IWPs and group cohomology at degree 3.

| Wyckoff position | Little group Intl. | Little group Schönflies | Coordinates $(0,0,0)+$ $(1/2,1/2,1/2)+$ | LSM anomaly class | Topo. inv. |
|---|---|---|---|---|---|
| 4a | $2/m$ | $C_{2h}$ | $(0,0,0), (0,1/2,0)$ | $A_c A_{c'}^2 + A_c A_{c'} A_i + A_{c'} A_i^2 + A_{c'} A_i A_{x+y+z} + A_{c'} B_{y(x+z)} + A_i B_{y(x+z)}$ | $\varphi_1[I]$ |
| 4b | $2/m$ | $C_{2h}$ | $(0,0,1/2), (0,1/2,1/2)$ | $A_c A_{c'}^2 + A_c A_{c'} A_i + A_i^3 + A_i^2 A_{x+y+z} + A_{c'} B_{y(x+z)} + A_i B_{y(x+z)}$ | $\varphi_1[T_1 T_2 I]$ |
| 4c | $2m$ | $C_{2h}$ | $(1/4,1/4,1/4), (3/4,1/4,1/4)$ | $(A_{c'} + A_i)(A_{c'} A_i + A_i A_{x+y+z} + B_{y(x+z)} + B_{z(x+y)})$ | $\varphi_1[T_1 T_2 T_3 I]$ |
| 4d | $2m$ | $C_{2h}$ | $(1/4,1/4,3/4), (3/4,1/4,3/4)$ | $(A_{c'} + A_i)(B_{y(x+z)} + B_{z(x+y)})$ | $\varphi_1[T_3 I]$ |
| 4e | $mm2$ | $C_{2v}$ | $(0,1/4,z), (0,3/4,-z)$ | $(A_c + A_{c'})B_{y(x+z)}$ | $\varphi_2[T_1 T_2, C_2]$ |

**No. 75:** *P4*

This group is generated by three translations $T_{1,2,3}$ as given in Eqs. (88), a two-fold rotation $C_2$, and a four-fold rotation $C_4$:

$$C_2 : (x, y, z) \rightarrow (-x, -y, z), \tag{363a}$$
$$C_4 : (x, y, z) \rightarrow (-y, x, z). \tag{363b}$$

The $\mathbb{Z}_2$ cohomology ring is given by

$$\mathbb{Z}_2[A_{\mathsf{q}}, A_{x+y}, A_z, B_\alpha, B_{xy}]/\langle \mathcal{R}_2, \mathcal{R}_3, \mathcal{R}_4 \rangle \tag{364}$$

where the relations are

$$\mathcal{R}_2 : \quad A_{\mathsf{q}} A_{x+y}, \quad A_{\mathsf{q}}^2, \quad A_z^2, \tag{365a}$$

$$\mathcal{R}_3 : \quad A_{x+y}^3 + A_{x+y} B_\alpha + A_{\mathsf{q}} B_{xy}, \quad A_{x+y}(A_{x+y}^2 + B_\alpha + B_{xy}), \tag{365b}$$

$$\mathcal{R}_4 : \quad B_{xy}(B_\alpha + B_{xy}). \tag{365c}$$

We have the following table regarding IWPs and group cohomology at degree 3.

| Wyckoff position | Little group Intl. | Little group Schönflies | Coordinates | LSM anomaly class | Topo. inv. |
|---|---|---|---|---|---|
| 1a | 4 | $C_4$ | $(0,0,z)$ | $A_z(A_{x+y}^2 + B_\alpha + B_{xy})$ | $\varphi_2[T_3, C_2]$ |
| 1b | 4 | $C_4$ | $(1/2,1/2,z)$ | $A_z B_{xy}$ | $\varphi_2[T_3, T_1 T_2 C_2]$ |
| 2c | 2 | $C_2$ | $(0,1/2,z), (1/2,0,z)$ | $A_{x+y}^2 A_z$ | $\varphi_2[T_3, T_1 C_2]$ |

**No. 76:** $P4_1$

This group is generated by three translations $T_{1,2,3}$ as given in Eqs. (88), a two-fold screw $S_2$, and a four-fold screw $S_4$:

$$S_2\colon (x,y,z) \to (-x,-y,z+1/2), \tag{366a}$$
$$S_4\colon (x,y,z) \to (-y,x,z+1/4). \tag{366b}$$

The $\mathbb{Z}_2$ cohomology ring is given by

$$\mathbb{Z}_2[A_{\mathsf{q}}, A_{x+y}, B_{xy}]/\langle \mathcal{R}_2, \mathcal{R}_3, \mathcal{R}_4 \rangle \tag{367}$$

where the relations are

$$\mathcal{R}_2\colon\quad A_{\mathsf{q}}A_{x+y},\quad A_{\mathsf{q}}^2, \tag{368a}$$
$$\mathcal{R}_3\colon\quad A_{x+y}^3 + A_{\mathsf{q}}B_{xy},\quad A_{x+y}(A_{x+y}^2 + B_{xy}), \tag{368b}$$
$$\mathcal{R}_4\colon\quad B_{xy}^2. \tag{368c}$$

We have the following table regarding IWPs and group cohomology at degree 3.

| Wyckoff position | Little group Intl. | Little group Schönflies | Coordinates | LSM anomaly class | Topo. inv. |
|---|---|---|---|---|---|
| 4a | 1 | $C_1$ | $(x,y,z)$, $(-x,-y,z+1/2)$, $(-y,x,z+1/4)$, $(y,-x,z+3/4)$ | $A_{x+y}^3$ | $\widehat{\varphi_3}[T_1, T_2, S_4]$ |

Here the topological invariant can be chosen to be

$$\begin{aligned}
\widehat{\varphi_3}[T_1, T_2, S_4] =&\, \lambda(T_1, T_2, T_1^{-1}S_4) + \lambda(T_2, T_1, T_1^{-1}S_4) + \lambda(T_1, T_1^{-1}S_4, T_1) + \lambda(T_2, S_4, T_2) \\
&+ \lambda(S_4, T_1, T_2) + \lambda(S_4, T_2, T_1).
\end{aligned} \tag{369}$$

**No. 77:** $P4_2$

This group is generated by three translations $T_{1,2,3}$ as given in Eqs. (88), a two-fold rotation $C_2$, and a four-fold screw $S_4$:

$$C_2\colon (x,y,z) \to (-x,-y,z), \tag{370a}$$
$$S_4\colon (x,y,z) \to (-y,x,z+1/2). \tag{370b}$$

The $\mathbb{Z}_2$ cohomology ring is given by

$$\mathbb{Z}_2[A_{\mathsf{q}}, A_{x+y}, A_z, B_{xy}]/\langle \mathcal{R}_2, \mathcal{R}_3, \mathcal{R}_4 \rangle \tag{371}$$

where the relations are

$$\mathcal{R}_2\colon\quad A_{\mathsf{q}}A_{x+y},\quad A_{\mathsf{q}}^2, \tag{372a}$$
$$\mathcal{R}_3\colon\quad A_{x+y}^3 + A_{x+y}A_z^2 + A_{\mathsf{q}}B_{xy},\quad A_{x+y}(A_{x+y}^2 + A_z^2 + B_{xy}), \tag{372b}$$
$$\mathcal{R}_4\colon\quad B_{xy}(A_z^2 + B_{xy}). \tag{372c}$$

We have the following table regarding IWPs and group cohomology at degree 3.

| Wyckoff position | Little group Intl. | Little group Schönflies | Coordinates | LSM anomaly class | Topo. inv. |
|---|---|---|---|---|---|
| 2a | 2 | $C_2$ | $(0,0,z)$, $(0,0,z+1/2)$ | $A_{x+y}^3 + A_{\mathsf{q}}A_z^2 + A_{x+y}A_z^2$ | $\varphi_2[S_4, C_2]$ |
| 2b | 2 | $C_2$ | $(1/2,1/2,z)$, $(1/2,1/2,z+1/2)$ | $A_{x+y}(A_{x+y}^2 + A_z^2)$ | $\varphi_2[T_1S_4, T_1T_2C_2]$ |
| 2c | 2 | $C_2$ | $(0,1/2,z)$, $(1/2,0,z+1/2)$ | $A_{x+y}^2(A_{x+y} + A_z)$ | $\varphi_2[T_3, T_1C_2]$ |

**No. 78:** $P4_3$

This group is generated by three translations $T_{1,2,3}$ as given in Eqs. (88), a two-fold screw $S_2$, and a four-fold screw $S_4$:

$$S_2 \colon (x, y, z) \to (-x, -y, z + 1/2), \tag{373a}$$
$$S_4 \colon (x, y, z) \to (-y, x, z + 3/4). \tag{373b}$$

The $\mathbb{Z}_2$ cohomology ring is given by

$$\mathbb{Z}_2[A_{\mathsf{q}}, A_{x+y}, B_{xy}]/\langle \mathcal{R}_2, \mathcal{R}_3, \mathcal{R}_4 \rangle \tag{374}$$

where the relations are

$$\mathcal{R}_2 \colon \quad A_{\mathsf{q}} A_{x+y}, \quad A_{\mathsf{q}}^2, \tag{375a}$$
$$\mathcal{R}_3 \colon \quad A_{x+y}^3 + A_{\mathsf{q}} B_{xy}, \quad A_{x+y}(A_{x+y}^2 + B_{xy}), \tag{375b}$$
$$\mathcal{R}_4 \colon \quad B_{xy}^2. \tag{375c}$$

We have the following table regarding IWPs and group cohomology at degree 3.

| Wyckoff position | Little group Intl. | Little group Schönflies | Coordinates | LSM anomaly class | Topo. inv. |
|---|---|---|---|---|---|
| 4a | 1 | $C_1$ | $(x, y, z), (-x, -y, z + 1/2),$ $(-y, x, z + 3/4), (y, -x, z + 1/4)$ | $A_{x+y}^3$ | $\widehat{\varphi_3}[T_1, T_2, S_4]$ |

Here the topological invariant $\widehat{\varphi_3}$ can be chosen to be the same as that of group No.76 $P4_1$, given by Eq. (369).

**No. 79:** $I4$

This group is generated by three translations $T_{1,2,3}$ as given in Eqs. (91), a two-fold rotation $C_2$, and a four-fold rotation $C_4$:

$$C_2 \colon (x, y, z) \to (-x, -y, z), \tag{376a}$$
$$C_4 \colon (x, y, z) \to (-y, x, z). \tag{376b}$$

The $\mathbb{Z}_2$ cohomology ring is given by

$$\mathbb{Z}_2[A_{\mathsf{q}}, A_{x+y+z}, B_\alpha, B_\beta, B_{z(x+y)}, C_\gamma, C_{xyz}]/\langle \mathcal{R}_2, \mathcal{R}_3, \mathcal{R}_4, \mathcal{R}_5, \mathcal{R}_6 \rangle \tag{377}$$

where the relations are

$$\mathcal{R}_2 \colon \quad A_{\mathsf{q}} A_{x+y+z}, \quad A_{\mathsf{q}}^2, \quad A_{x+y+z}^2, \tag{378a}$$
$$\mathcal{R}_3 \colon \quad A_{x+y+z} B_\alpha + A_{\mathsf{q}} B_\beta, \quad A_{x+y+z}(B_\alpha + B_\beta), \quad A_{x+y+z} B_\alpha + A_{\mathsf{q}} B_{z(x+y)}, \quad A_{x+y+z}(B_\alpha + B_{z(x+y)}), \tag{378b}$$
$$\mathcal{R}_4 \colon \quad A_{\mathsf{q}} C_\gamma + A_{x+y+z} C_\gamma + A_{\mathsf{q}} C_{xyz}, \quad A_{x+y+z} C_{xyz}, \quad B_\alpha B_\beta + B_\alpha B_{z(x+y)} + A_{\mathsf{q}} C_\gamma, \quad B_\alpha B_\beta + B_\beta^2 + A_{\mathsf{q}} C_\gamma,$$
$$\quad B_\alpha B_\beta + B_\beta B_{z(x+y)} + A_{\mathsf{q}} C_\gamma + A_{x+y+z} C_\gamma, \quad B_\alpha B_\beta + B_{z(x+y)}^2 + A_{\mathsf{q}} C_\gamma, \tag{378c}$$
$$\mathcal{R}_5 \colon \quad A_{x+y+z} B_\alpha^2 + B_\beta C_\gamma + B_{z(x+y)} C_\gamma, \quad A_{x+y+z} B_\alpha^2 + B_\alpha C_\gamma + B_\beta C_\gamma + B_\alpha C_{xyz}, \quad B_\beta C_{xyz}, \quad B_{z(x+y)} C_{xyz}, \tag{378d}$$
$$\mathcal{R}_6 \colon \quad B_\alpha^2 B_\beta + A_{\mathsf{q}} B_\alpha C_\gamma + C_\gamma^2, \quad C_\gamma C_{xyz}, \quad C_{xyz}^2. \tag{378e}$$

We have the following table regarding IWPs and group cohomology at degree 3.

| Wyckoff position | Little group Intl. | Little group Schönflies | Coordinates $(0,0,0)+\ (1/2,1/2,1/2)+$ | LSM anomaly class | Topo. inv. |
|---|---|---|---|---|---|
| 2a | 4 | $C_4$ | $(0, 0, z)$ | $C_{xyz}$ | $\varphi_2[T_1 T_2, C_2]$ |
| 4b | 2 | $C_2$ | $(0, 1/2, z), (1/2, 0, z)$ | $A_{x+y+z} B_\alpha$ | $\varphi_2[T_2 C_4, T_2 T_3 C_2]$ |

**No. 80:** $I4_1$

This group is generated by three translations $T_{1,2,3}$ as given in Eqs. (91), a two-fold rotation $C_2$, and a four-fold screw $S_4$:

$$C_2 \colon (x, y, z) \to (-x, -y, z), \tag{379a}$$
$$S_4 \colon (x, y, z) \to (-y - 1/2, x, z + 3/4). \tag{379b}$$

The $\mathbb{Z}_2$ cohomology ring is given by

$$\mathbb{Z}_2[A_{\mathsf{q}}, A_{x+y+z}, B_{z(x+y)}, C_\gamma]/\langle \mathcal{R}_2, \mathcal{R}_3, \mathcal{R}_4, \mathcal{R}_5, \mathcal{R}_6 \rangle \tag{380}$$

where the relations are

$$\mathcal{R}_2 \colon \quad A_{\mathsf{q}} A_{x+y+z}, \quad A_{\mathsf{q}}^2, \tag{381a}$$
$$\mathcal{R}_3 \colon \quad A_{\mathsf{q}} B_{z(x+y)}, \quad A_{x+y+z} B_{z(x+y)}, \tag{381b}$$
$$\mathcal{R}_4 \colon \quad A_{\mathsf{q}} C_\gamma, \quad B_{z(x+y)}^2, \tag{381c}$$
$$\mathcal{R}_5 \colon \quad B_{z(x+y)} C_\gamma, \tag{381d}$$
$$\mathcal{R}_6 \colon \quad C_\gamma^2. \tag{381e}$$

We have the following table regarding IWPs and group cohomology at degree 3.

| Wyckoff position | Little group Intl. | Little group Schönflies | Coordinates $(0,0,0)+$ $(1/2,1/2,1/2)+$ | LSM anomaly class | Topo. inv. |
|---|---|---|---|---|---|
| 4a | 2 | $C_2$ | $(0,0,z)$, $(0,1/2,z+1/4)$ | $C_\gamma$ | $\varphi_2[T_1 T_2, C_2]$ |

**No. 81:** $P\overline{4}$

This group is generated by three translations $T_{1,2,3}$ as given in Eqs. (88), a two-fold rotation $C_2$, and a four-fold roto-inversion $\overline{C}_4$:

$$C_2 \colon (x, y, z) \to (-x, -y, z), \tag{382a}$$
$$\overline{C}_4 \colon (x, y, z) \to (y, -x, -z). \tag{382b}$$

The $\mathbb{Z}_2$ cohomology ring is given by

$$\mathbb{Z}_2[A_{\mathsf{q}}, A_{x+y}, A_z, B_\alpha, B_{xy}]/\langle \mathcal{R}_2, \mathcal{R}_3, \mathcal{R}_4 \rangle \tag{383}$$

where the relations are

$$\mathcal{R}_2 \colon \quad A_{\mathsf{q}} A_{x+y}, \quad A_{\mathsf{q}}^2, \quad A_z(A_{\mathsf{q}} + A_z), \tag{384a}$$
$$\mathcal{R}_3 \colon \quad A_{x+y}^3 + A_{x+y} B_\alpha + A_{\mathsf{q}} B_{xy}, \quad A_{x+y}(A_{x+y}^2 + B_\alpha + B_{xy}), \tag{384b}$$
$$\mathcal{R}_4 \colon \quad B_{xy}(B_\alpha + B_{xy}). \tag{384c}$$

We have the following table regarding IWPs and group cohomology at degree 3.

| Wyckoff position | Little group Intl. | Little group Schönflies | Coordinates | LSM anomaly class | Topo. inv. |
|---|---|---|---|---|---|
| 1a | $\overline{4}$ | $S_4$ | $(0,0,0)$ | $(A_{\mathsf{q}} + A_z)(A_{x+y}^2 + B_\alpha + B_{xy})$ | $\varphi_2[\overline{C}_4, C_2]$ |
| 1b | $\overline{4}$ | $S_4$ | $(0,0,1/2)$ | $A_z(A_{x+y}^2 + B_\alpha + B_{xy})$ | $\varphi_2[T_3\overline{C}_4, C_2]$ |
| 1c | $\overline{4}$ | $S_4$ | $(1/2,1/2,0)$ | $A_{x+y}^3 + A_{x+y}B_\alpha + A_z B_{xy}$ | $\varphi_2[T_2\overline{C}_4, T_1 T_2 C_2]$ |
| 1d | $\overline{4}$ | $S_4$ | $(1/2,1/2,1/2)$ | $A_z B_{xy}$ | $\varphi_2[T_2 T_3\overline{C}_4, T_1 T_2 C_2]$ |
| 2g | 2 | $C_2$ | $(0,1/2,z)$, $(1/2,0,-z)$ | $A_{x+y}^2 A_z$ | $\varphi_2[T_3, T_1 C_2]$ |

**No. 82:** $I\bar{4}$

This group is generated by three translations $T_{1,2,3}$ as given in Eqs. (91), a two-fold rotation $C_2$, and a four-fold roto-inversion $\overline{C}_4$:

$$C_2 \colon (x, y, z) \to (-x, -y, z), \tag{385a}$$

$$\overline{C}_4 \colon (x, y, z) \to (y, -x, -z). \tag{385b}$$

The $\mathbb{Z}_2$ cohomology ring is given by

$$\mathbb{Z}_2[A_{\mathsf{q}}, A_{x+y+z}, B_\alpha, B_\beta, B_{z(x+y)}, C_\gamma, C_{xyz}]/\langle \mathscr{R}_2, \mathscr{R}_3, \mathscr{R}_4, \mathscr{R}_5, \mathscr{R}_6 \rangle \tag{386}$$

where the relations are

$$\mathscr{R}_2\colon \quad A_{\mathsf{q}}A_{x+y+z}, \quad A_{\mathsf{q}}^2, \quad A_{x+y+z}^2, \tag{387a}$$

$$\mathscr{R}_3\colon \quad A_{x+y+z}B_\alpha + A_{\mathsf{q}}B_\beta, \quad A_{x+y+z}(B_\alpha + B_\beta), \quad A_{x+y+z}B_\alpha + A_{\mathsf{q}}B_{z(x+y)}, \quad A_{x+y+z}(B_\alpha + B_{z(x+y)}), \tag{387b}$$

$$\mathscr{R}_4\colon \quad A_{\mathsf{q}}C_\gamma + A_{x+y+z}C_\gamma + A_{\mathsf{q}}C_{xyz}, \quad A_{x+y+z}C_{xyz}, \quad B_\alpha B_\beta + B_\alpha B_{z(x+y)} + A_{\mathsf{q}}C_\gamma,$$
$$B_\alpha B_\beta + B_\beta^2 + A_{\mathsf{q}}C_\gamma, \quad B_\alpha B_\beta + B_\beta B_{z(x+y)} + A_{\mathsf{q}}C_\gamma + A_{x+y+z}C_\gamma, \quad B_\alpha B_\beta + B_{z(x+y)}^2 + A_{\mathsf{q}}C_\gamma, \tag{387c}$$

$$\mathscr{R}_5\colon \quad (B_\beta + B_{z(x+y)})C_\gamma, \quad B_\alpha C_\gamma + B_\beta C_\gamma + B_\alpha C_{xyz}, \quad B_\beta C_{xyz}, \quad B_{z(x+y)}C_{xyz}, \tag{387d}$$

$$\mathscr{R}_6\colon \quad C_\gamma(A_{\mathsf{q}}B_\alpha + C_\gamma), \quad C_\gamma(A_{\mathsf{q}}B_\alpha + A_{x+y+z}B_\alpha + C_{xyz}), \quad A_{\mathsf{q}}B_\alpha C_\gamma + A_{x+y+z}B_\alpha C_\gamma + C_{xyz}^2. \tag{387e}$$

We have the following table regarding IWPs and group cohomology at degree 3.

| Wyckoff position | Little group Intl. | Schönflies | Coordinates $(0,0,0)+$ $(1/2,1/2,1/2)+$ | LSM anomaly class | Topo. inv. |
|---|---|---|---|---|---|
| 2a | $\bar{4}$ | $S_4$ | $(0,0,0)$ | $A_{\mathsf{q}}B_\alpha + A_{x+y+z}B_\alpha + C_{xyz}$ | $\varphi_2[\overline{C}_4, C_2]$ |
| 2b | $\bar{4}$ | $S_4$ | $(0,0,1/2)$ | $C_{xyz}$ | $\varphi_2[T_1 T_2 \overline{C}_4, C_2]$ |
| 2c | $\bar{4}$ | $S_4$ | $(0,1/2,1/4)$ | $A_{x+y+z}B_\alpha + C_\gamma + C_{xyz}$ | $\varphi_2[T_1 \overline{C}_4, T_1 T_3 C_2]$ |
| 2d | $\bar{4}$ | $S_4$ | $(0,1/2,3/4)$ | $C_\gamma + C_{xyz}$ | $\varphi_2[T_2^{-1}\overline{C}_4, T_1 T_3 C_2]$ |

**No. 83:** $P4/m$

This group is generated by three translations $T_{1,2,3}$ as given in Eqs. (88), a two-fold rotation $C_2$, a four-fold rotation $C_4$, and an inversion $I$:

$$C_2 \colon (x, y, z) \to (-x, -y, z), \tag{388a}$$

$$C_4 \colon (x, y, z) \to (-y, x, z), \tag{388b}$$

$$I \colon (x, y, z) \to (-x, -y, -z). \tag{388c}$$

The $\mathbb{Z}_2$ cohomology ring is given by

$$\mathbb{Z}_2[A_i, A_{\mathsf{q}}, A_{x+y}, A_z, B_\alpha, B_{xy}]/\langle \mathscr{R}_2, \mathscr{R}_3, \mathscr{R}_4 \rangle \tag{389}$$

where the relations are

$$\mathscr{R}_2\colon \quad A_{\mathsf{q}}A_{x+y}, \quad A_{\mathsf{q}}^2, \quad A_z(A_i + A_z), \tag{390a}$$

$$\mathscr{R}_3\colon \quad A_{x+y}^3 + A_{x+y}B_\alpha + A_{\mathsf{q}}B_{xy}, \quad A_{x+y}(A_{x+y}^2 + B_\alpha + B_{xy}), \tag{390b}$$

$$\mathscr{R}_4\colon \quad B_{xy}(B_\alpha + B_{xy}). \tag{390c}$$

We have the following table regarding IWPs and group cohomology at degree 3.

| Wyckoff position | Little group Intl. | Schönflies | Coordinates | LSM anomaly class | Topo. inv. |
|---|---|---|---|---|---|
| 1a | $4/m$ | $C_{4h}$ | $(0,0,0)$ | $(A_i + A_z)(A_{x+y}^2 + B_\alpha + B_{xy})$ | $\varphi_1[I]$ |
| 1b | $4/m$ | $C_{4h}$ | $(0,0,1/2)$ | $A_z(A_{x+y}^2 + B_\alpha + B_{xy})$ | $\varphi_1[T_3 I]$ |
| 1c | $4/m$ | $C_{4h}$ | $(1/2,1/2,0)$ | $(A_i + A_z)B_{xy}$ | $\varphi_1[T_1 T_2 I]$ |
| 1d | $4/m$ | $C_{4h}$ | $(1/2,1/2,1/2)$ | $A_z B_{xy}$ | $\varphi_1[T_1 T_2 T_3 I]$ |
| 2e | $2/m$ | $C_{2h}$ | $(0,1/2,0), (1/2,0,0)$ | $A_{x+y}^2(A_i + A_z)$ | $\varphi_1[T_1 I]$ |
| 2f | $2/m$ | $C_{2h}$ | $(0,1/2,1/2), (1/2,0,1/2)$ | $A_{x+y}^2 A_z$ | $\varphi_1[T_1 T_3 I]$ |

**No. 84: $P4_2/m$**

This group is generated by three translations $T_{1,2,3}$ as given in Eqs. (88), a two-fold rotation $C_2$, a four-fold screw $S_4$, and an inversion $I$:

$$C_2: (x, y, z) \rightarrow (-x, -y, z), \tag{391a}$$
$$S_4: (x, y, z) \rightarrow (-y, x, z + 1/2), \tag{391b}$$
$$I: (x, y, z) \rightarrow (-x, -y, -z). \tag{391c}$$

The $\mathbb{Z}_2$ cohomology ring is given by

$$\mathbb{Z}_2[A_i, A_{\mathsf{q}}, A_{x+y}, B_\alpha, B_\beta, B_{xy}, B_{z(x+y)}]/\langle \mathcal{R}_2, \mathcal{R}_3, \mathcal{R}_4 \rangle \tag{392}$$

where the relations are

$$\mathcal{R}_2: \quad A_i A_{\mathsf{q}}, \quad A_{\mathsf{q}} A_{x+y}, \quad A_{\mathsf{q}}^2, \tag{393a}$$
$$\mathcal{R}_3: \quad A_{\mathsf{q}} B_\beta, \quad A_{x+y}(A_i^2 + A_i A_{x+y} + B_\beta), \quad A_{x+y}^3 + A_{x+y} B_\alpha + A_{\mathsf{q}} B_{xy}, \quad A_{x+y}(A_{x+y}^2 + B_\alpha + B_{xy}), \quad A_{\mathsf{q}} B_{z(x+y)}, \tag{393b}$$
$$\mathcal{R}_4: \quad A_i^4 + A_i^2 B_\alpha + B_\beta^2, \quad A_i^2 B_{xy} + B_\beta B_{xy} + A_{x+y}^2 B_{z(x+y)} + B_\alpha B_{z(x+y)},$$
$$A_i^2 B_{xy} + A_i^2 B_{z(x+y)} + A_i A_{x+y} B_{z(x+y)} + B_\beta B_{z(x+y)}, \quad B_{xy}(B_\alpha + B_{xy}),$$
$$(A_{x+y}^2 + B_\alpha + B_{xy}) B_{z(x+y)}, \quad A_i A_{x+y}^3 + A_{x+y}^4 + A_i^2 B_{xy} + A_i A_{x+y} B_{z(x+y)} + B_{z(x+y)}^2. \tag{393c}$$

We have the following table regarding IWPs and group cohomology at degree 3.

| Wyckoff position | Little group Intl. | Little group Schönflies | Coordinates | LSM anomaly class | Topo. inv. |
|---|---|---|---|---|---|
| 2a | $2/m$ | $C_{2h}$ | $(0,0,0), (0,0,1/2)$ | $(A_i + A_{\mathsf{q}})(A_{x+y}^2 + B_\alpha + B_{xy})$ | $\varphi_1[I]$ |
| 2b | $2/m$ | $C_{2h}$ | $(1/2,1/2,0), (1/2,1/2,1/2)$ | $A_{x+y}^3 + A_{x+y} B_\alpha + A_i B_{xy}$ | $\varphi_1[T_1 T_2 I]$ |
| 2c | $2/m$ | $C_{2h}$ | $(0,1/2,0), (1/2,0,1/2)$ | $A_{x+y}(A_i A_{x+y} + A_{x+y}^2 + B_{z(x+y)})$ | $\varphi_1[T_2 I]$ |
| 2d | $2/m$ | $C_{2h}$ | $(0,1/2,1/2), (1/2,0,0)$ | $A_{x+y}(A_{x+y}^2 + B_{z(x+y)})$ | $\varphi_1[T_1 I]$ |
| 2e | $\bar{4}$ | $S_4$ | $(0,0,1/4), (0,0,3/4)$ | $A_{x+y}^3 + A_{\mathsf{q}} B_\alpha + A_{x+y} B_\alpha$ | $\varphi_2[S_4 I, C_2]$ |
| 2f | $\bar{4}$ | $S_4$ | $(1/2,1/2,1/4), (1/2,1/2,3/4)$ | $A_{x+y}(A_{x+y}^2 + B_\alpha)$ | $\varphi_2[T_2 S_4 I, T_1 T_2 C_2]$ |

**No. 85: $P4/n$**

This group is generated by three translations $T_{1,2,3}$ as given in Eqs. (88), a two-fold rotation $C_2$, a four-fold rotation $C_4$, and an inversion $I$:

$$C_2: (x, y, z) \rightarrow (-x + 1/2, -y + 1/2, z), \tag{394a}$$
$$C_4: (x, y, z) \rightarrow (-y + 1/2, x, z), \tag{394b}$$
$$I: (x, y, z) \rightarrow (-x, -y, -z). \tag{394c}$$

The $\mathbb{Z}_2$ cohomology ring is given by

$$\mathbb{Z}_2[A_i, A_{\mathsf{q}}, A_z, B_\alpha, B_\beta]/\langle \mathcal{R}_2, \mathcal{R}_3, \mathcal{R}_4 \rangle \tag{395}$$

where the relations are

$$\mathcal{R}_2: \quad A_i A_{\mathsf{q}}, \quad A_{\mathsf{q}}^2, \quad A_z(A_i + A_z), \tag{396a}$$
$$\mathcal{R}_3: \quad A_i(B_\alpha + B_\beta), \quad A_i^3 + A_i B_\alpha + A_{\mathsf{q}} B_\beta, \tag{396b}$$
$$\mathcal{R}_4: \quad B_\beta(B_\alpha + B_\beta). \tag{396c}$$

We have the following table regarding IWPs and group cohomology at degree 3.

| Wyckoff position | Little group Intl. | Little group Schönflies | Coordinates | LSM anomaly class | Topo. inv. |
|---|---|---|---|---|---|
| 2a | $\bar{4}$ | $S_4$ | $(1/4, 3/4, 0), (3/4, 1/4, 0)$ | $A_i^3 + A_i^2 A_z + A_i B_\alpha + A_z B_\beta$ | $\varphi_2[C_4 I, T_2^{-1} C_2]$ |
| 2b | $\bar{4}$ | $S_4$ | $(1/4, 3/4, 1/2), (3/4, 1/4, 1/2)$ | $A_z(A_i^2 + B_\beta)$ | $\varphi_2[T_3 C_4 I, T_2^{-1} C_2]$ |
| 2c | 4 | $C_4$ | $(1/4, 1/4, z), (3/4, 3/4, -z)$ | $A_z(B_\alpha + B_\beta)$ | $\varphi_2[T_3, C_2]$ |
| 4d | $\bar{1}$ | $C_i$ | $(0,0,0), (1/2,1/2,0),$ $(1/2,0,0), (0,1/2,0)$ | $A_i^2(A_i + A_z)$ | $\varphi_1[I]$ |
| 4e | $\bar{1}$ | $C_i$ | $(0,0,1/2), (1/2,1/2,1/2),$ $(1/2,0,1/2), (0,1/2,1/2)$ | $A_i^2 A_z$ | $\varphi_1[T_3 I]$ |

**No. 86:** $P4_2/n$

This group is generated by three translations $T_{1,2,3}$ as given in Eqs. (88), a two-fold screw $S_2$, a four-fold screw $S_4$, and an inversion $I$:

$$S_2: (x,y,z) \to (-x - 1/2, -y + 1/2, z + 1), \tag{397a}$$

$$S_4: (x,y,z) \to (-y, x + 1/2, z + 1/2), \tag{397b}$$

$$I: (x,y,z) \to (-x, -y, -z). \tag{397c}$$

The $\mathbb{Z}_2$ cohomology ring is given by

$$\mathbb{Z}_2[A_i, A_\mathsf{q}, A_{x+y+z}, B_\alpha]/\langle \mathcal{R}_2, \mathcal{R}_3, \mathcal{R}_4 \rangle \tag{398}$$

where the relations are

$$\mathcal{R}_2: \quad A_i A_\mathsf{q}, \quad A_\mathsf{q}^2, \tag{399a}$$

$$\mathcal{R}_3: \quad A_i A_{x+y+z}(A_i + A_{x+y+z}), \quad A_i^3 + A_\mathsf{q} A_{x+y+z}^2 + A_i B_\alpha + A_\mathsf{q} B_\alpha, \tag{399b}$$

$$\mathcal{R}_4: \quad A_{x+y+z}(A_i + A_{x+y+z})(A_{x+y+z}^2 + B_\alpha). \tag{399c}$$

We have the following table regarding IWPs and group cohomology at degree 3.

| Wyckoff position | Little group Intl. | Little group Schönflies | Coordinates | LSM anomaly class | Topo. inv. |
|---|---|---|---|---|---|
| 2a | $\bar{4}$ | $S_4$ | $(1/4, 1/4, 1/4), (3/4, 3/4, 3/4)$ | $(A_i + A_{x+y+z})(A_i^2 + A_i A_{x+y+z} + A_{x+y+z}^2 + B_\alpha)$ | $\varphi_2[S_4 I, T_1 T_3^{-1} S_2]$ |
| 2b | $\bar{4}$ | $S_4$ | $(1/4, 1/4, 3/4), (3/4, 3/4, 1/4)$ | $A_{x+y+z}(A_{x+y+z}^2 + B_\alpha)$ | $\varphi_2[T_3 S_4 I, T_1 T_3^{-1} S_2]$ |
| 4c | $\bar{1}$ | $C_i$ | $(0,0,0), (1/2,1/2,0),$ $(0,1/2,1/2), (1/2,0,1/2)$ | $A_i^2(A_i + A_{x+y+z})$ | $\varphi_1[I]$ |
| 4d | $\bar{1}$ | $C_i$ | $(0,0,1/2), (1/2,1/2,1/2),$ $(0,1/2,0), (1/2,0,0)$ | $A_i^2 A_{x+y+z}$ | $\varphi_1[T_3 I]$ |
| 4e | 2 | $C_2$ | $(3/4, 1/4, z), (3/4, 1/4, z+1/2),$ $(1/4, 3/4, -z), (1/4, 3/4, -z+1/2)$ | $A_\mathsf{q} A_{x+y+z}^2$ | $\varphi_2[S_4, T_3^{-1} S_2]$ |

**No. 87:** $I4/m$

This group is generated by three translations $T_{1,2,3}$ as given in Eqs. (91), a two-fold rotation $C_2$, a four-fold rotation $C_4$, and an inversion $I$:

$$C_2: (x,y,z) \to (-x, -y, z), \tag{400a}$$

$$C_4: (x,y,z) \to (-y, x, z), \tag{400b}$$

$$I: (x,y,z) \to (-x, -y, -z). \tag{400c}$$

The $\mathbb{Z}_2$ cohomology ring is given by

$$\mathbb{Z}_2[A_i, A_\mathsf{q}, A_{x+y+z}, B_\alpha, B_\beta, B_{z(x+y)}, C_{xyz}]/\langle \mathcal{R}_2, \mathcal{R}_3, \mathcal{R}_4, \mathcal{R}_5, \mathcal{R}_6 \rangle \tag{401}$$

where the relations are

$$\mathcal{R}_2: \quad A_{\mathsf{q}}A_{x+y+z}, \quad A_{\mathsf{q}}^2, \quad A_{x+y+z}(A_i + A_{x+y+z}), \tag{402a}$$

$$\mathcal{R}_3: \quad A_i^2 A_{x+y+z} + A_{x+y+z}B_\alpha + A_{\mathsf{q}}B_\beta, \quad A_{x+y+z}(A_i^2 + B_\alpha + B_\beta), \quad A_i^2 A_{x+y+z} + A_{x+y+z}B_\alpha + A_{\mathsf{q}}B_{z(x+y)},$$

$$A_{x+y+z}(A_i^2 + B_\alpha + B_{z(x+y)}), \tag{402b}$$

$$\mathcal{R}_4: \quad A_{x+y+z}C_{xyz}, \quad B_\beta^2 + A_i^2 B_{z(x+y)} + B_\alpha B_{z(x+y)}, \quad B_\alpha B_\beta + B_\beta B_{z(x+y)} + A_{\mathsf{q}}C_{xyz}, \quad B_{z(x+y)}(B_\alpha + B_{z(x+y)}), \tag{402c}$$

$$\mathcal{R}_5: \quad (A_i A_{\mathsf{q}} + B_\beta)C_{xyz}, \quad B_{z(x+y)}C_{xyz}, \tag{402d}$$

$$\mathcal{R}_6: \quad C_{xyz}(A_i B_\alpha + C_{xyz}). \tag{402e}$$

We have the following table regarding IWPs and group cohomology at degree 3.

| Wyckoff position | Little group Intl. | Little group Schönflies | Coordinates $(0,0,0)+$ $(1/2,1/2,1/2)+$ | LSM anomaly class | Topo. inv. |
|---|---|---|---|---|---|
| 2a | $4/m$ | $C_{4h}$ | $(0,0,0)$ | $A_i^2 A_{x+y+z} + A_i B_\alpha + A_i B_{z(x+y)} + C_{xyz}$ | $\varphi_1[I]$ |
| 2b | $4/m$ | $C_{4h}$ | $(0,0,1/2)$ | $C_{xyz}$ | $\varphi_1[T_1 T_2 I]$ |
| 4c | $2/m$ | $C_{2h}$ | $(0,1/2,0), (1/2,0,0)$ | $A_i^2 A_{x+y+z} + A_{x+y+z}B_\alpha + A_i B_{z(x+y)}$ | $\varphi_1[T_2 T_3 I]$ |
| 4d | $\bar{4}$ | $S_4$ | $(0,1/2,1/4), (1/2,0,1/4)$ | $A_{x+y+z}(A_i^2 + B_\alpha)$ | $\varphi_2[T_1 C_4 I, T_1 T_3 C_2]$ |
| 8f | $\bar{1}$ | $C_i$ | $(1/4,1/4,1/4), (3/4,3/4,1/4),$ $(3/4,1/4,1/4), (1/4,3/4,1/4)$ | $A_i^2 A_{x+y+z}$ | $\varphi_1[T_1 T_2 T_3 I]$ |

**No. 88:** $I4_1/a$

This group is generated by three translations $T_{1,2,3}$ as given in Eqs. (91), a two-fold rotation $C_2$, a four-fold screw $S_4$, and an inversion $I$:

$$C_2: (x,y,z) \to (-x, -y+1/2, z), \tag{403a}$$

$$S_4: (x,y,z) \to (-y+3/4, x+1/4, z+1/4), \tag{403b}$$

$$I: (x,y,z) \to (-x,-y,-z). \tag{403c}$$

The $\mathbb{Z}_2$ cohomology ring is given by

$$\mathbb{Z}_2[A_i, A_{\mathsf{q}}, B_\alpha, B_{z(x+y)}, C_\gamma]/\langle \mathcal{R}_2, \mathcal{R}_3, \mathcal{R}_4, \mathcal{R}_5, \mathcal{R}_6 \rangle \tag{404}$$

where the relations are

$$\mathcal{R}_2: \quad A_i A_{\mathsf{q}}, \quad A_{\mathsf{q}}^2, \tag{405a}$$

$$\mathcal{R}_3: \quad A_i^3 + A_i B_\alpha + A_{\mathsf{q}}B_\alpha, \quad A_i^3 + A_i B_\alpha + A_{\mathsf{q}}B_{z(x+y)}, \tag{405b}$$

$$\mathcal{R}_4: \quad (A_i + A_{\mathsf{q}})C_\gamma, \quad A_i^4 + B_\alpha^2 + A_i^2 B_{z(x+y)} + B_\alpha B_{z(x+y)} + A_i C_\gamma, \quad A_i^4 + B_\alpha^2 + A_i^2 B_{z(x+y)} + B_{z(x+y)}^2, \tag{405c}$$

$$\mathcal{R}_5: \quad (B_\alpha + B_{z(x+y)})C_\gamma, \tag{405d}$$

$$\mathcal{R}_6: \quad C_\gamma(A_i B_\alpha + C_\gamma). \tag{405e}$$

We have the following table regarding IWPs and group cohomology at degree 3.

| Wyckoff position | Little group Intl. | Little group Schönflies | Coordinates $(0,0,0)+$ $(1/2,1/2,1/2)+$ | LSM anomaly class | Topo. inv. |
|---|---|---|---|---|---|
| 4a | $\bar{4}$ | $S_4$ | $(0,1/4,1/8), (1/2,1/4,3/8)$ | $A_i^3 + A_i B_\alpha + C_\gamma$ | $\varphi_2[S_4 I, T_1^{-1}T_2 C_2]$ |
| 4b | $\bar{4}$ | $S_4$ | $(0,1/4,5/8), (1/2,1/4,7/8)$ | $C_\gamma$ | $\varphi_2[T_1 T_2 S_4 I, T_1^{-1}T_2 C_2]$ |
| 8c | $\bar{1}$ | $C_i$ | $(0,0,0), (1/2,0,1/2),$ $(3/4,1/4,1/4), (3/4,3/4,3/4)$ | $A_i(A_i^2 + B_\alpha + B_{z(x+y)})$ | $\varphi_1[I]$ |
| 8d | $\bar{1}$ | $C_i$ | $(0,0,1/2), (1/2,0,0),$ $(3/4,1/4,3/4), (3/4,3/4,1/4)$ | $A_i(B_\alpha + B_{z(x+y)})$ | $\varphi_1[T_1 T_2 I]$ |

**No. 89:** $P422$

This group is generated by three translations $T_{1,2,3}$ as given in Eqs. (88), a two-fold rotation $C_2$, a two-fold rotation $C_2'$, and a two-fold rotation $C_2''$:

$$C_2 \colon (x, y, z) \to (-x, -y, z), \tag{406a}$$
$$C_2' \colon (x, y, z) \to (-x, y, -z), \tag{406b}$$
$$C_2'' \colon (x, y, z) \to (-y, -x, -z). \tag{406c}$$

The $\mathbb{Z}_2$ cohomology ring is given by

$$\mathbb{Z}_2[A_{c'}, A_{c''}, A_{x+y}, A_z, B_\alpha, B_{xy}]/\langle \mathcal{R}_2, \mathcal{R}_3, \mathcal{R}_4 \rangle \tag{407}$$

where the relations are

$$\mathcal{R}_2 \colon \quad A_{c'} A_{c''}, \quad A_{c''} A_{x+y}, \quad A_z(A_{c'} + A_{c''} + A_z), \tag{408a}$$
$$\mathcal{R}_3 \colon \quad A_{c'} A_{x+y}^2 + A_{x+y}^3 + A_{x+y} B_\alpha + A_{c'} B_{xy}, \quad A_{x+y}(A_{c'} A_{x+y} + A_{x+y}^2 + B_\alpha + B_{xy}), \tag{408b}$$
$$\mathcal{R}_4 \colon \quad B_{xy}(B_\alpha + B_{xy}). \tag{408c}$$

We have the following table regarding IWPs and group cohomology at degree 3.

| Wyckoff position | Little group Intl. | Little group Schönflies | Coordinates | LSM anomaly class | Topo. inv. |
|---|---|---|---|---|---|
| 1a | 422 | $D_4$ | $(0,0,0)$ | $(A_{c'} + A_{c''} + A_z)(A_{c'} A_{x+y} + A_{x+y}^2 + B_\alpha + B_{xy})$ | $\varphi_2[C_2, C_2']$ |
| 1b | 422 | $D_4$ | $(0,0,1/2)$ | $A_z(A_{c'} A_{x+y} + A_{x+y}^2 + B_\alpha + B_{xy})$ | $\varphi_2[C_2, T_3 C_2']$ |
| 1c | 422 | $D_4$ | $(1/2,1/2,0)$ | $(A_{c'} + A_{c''} + A_z)B_{xy}$ | $\varphi_2[T_1 T_2 C_2, T_1 C_2']$ |
| 1d | 422 | $D_4$ | $(1/2,1/2,1/2)$ | $A_z B_{xy}$ | $\varphi_2[T_1 T_2 C_2, T_1 T_3 C_2']$ |
| 2e | 222 | $D_2$ | $(1/2,0,0), (0,1/2,0)$ | $A_{x+y}(A_{c'} + A_{x+y})(A_{c'} + A_z)$ | $\varphi_2[T_1 C_2, T_1 C_2']$ |
| 2f | 222 | $D_2$ | $(1/2,0,1/2), (0,1/2,1/2)$ | $A_{x+y}(A_{c'} + A_{x+y})A_z$ | $\varphi_2[T_1 C_2, T_1 T_3 C_2']$ |

**No. 90:** $P42_12$

This group is generated by three translations $T_{1,2,3}$ as given in Eqs. (88), a two-fold rotation $C_2$, a two-fold screw $S_2'$, and a two-fold rotation $C_2''$:

$$C_2 \colon (x, y, z) \to (-x, -y, z), \tag{409a}$$
$$S_2' \colon (x, y, z) \to (-x + 1/2, y + 1/2, -z), \tag{409b}$$
$$C_2'' \colon (x, y, z) \to (-y, -x, -z). \tag{409c}$$

The $\mathbb{Z}_2$ cohomology ring is given by

$$\mathbb{Z}_2[A_{c'}, A_{c''}, A_z, B_\alpha, B_\beta, C_\gamma]/\langle \mathcal{R}_2, \mathcal{R}_3, \mathcal{R}_4, \mathcal{R}_5, \mathcal{R}_6 \rangle \tag{410}$$

where the relations are

$$\mathcal{R}_2 \colon \quad A_{c'} A_{c''}, \quad A_{c'}^2, \quad A_z(A_{c'} + A_{c''} + A_z), \tag{411a}$$
$$\mathcal{R}_3 \colon \quad A_{c'}(B_\alpha + B_\beta), \quad A_{c'} B_\alpha + A_{c''} B_\beta, \tag{411b}$$
$$\mathcal{R}_4 \colon \quad A_{c'} C_\gamma, \quad B_\beta(B_\alpha + B_\beta), \tag{411c}$$
$$\mathcal{R}_5 \colon \quad B_\beta C_\gamma, \tag{411d}$$
$$\mathcal{R}_6 \colon \quad B_\alpha^3 + B_\alpha^2 B_\beta + A_{c''} B_\alpha C_\gamma + C_\gamma^2. \tag{411e}$$

We have the following table regarding IWPs and group cohomology at degree 3.

| Wyckoff position | Little group Intl. | Little group Schönflies | Coordinates | LSM anomaly class | Topo. inv. |
|---|---|---|---|---|---|
| 2a | 222 | $D_2$ | $(0,0,0), (1/2,1/2,0)$ | $(A_{c'} + A_{c''} + A_z)B_\alpha + A_z B_\beta$ | $\varphi_2[C_2, C_2'']$ |
| 2b | 222 | $D_2$ | $(0,0,1/2), (1/2,1/2,1/2)$ | $A_z(B_\alpha + B_\beta)$ | $\varphi_2[C_2, T_3 C_2'']$ |
| 2c | 4 | $C_4$ | $(0,1/2,z), (1/2,0,-z)$ | $A_z B_\beta$ | $\varphi_2[T_3, T_1 C_2]$ |

**No. 91:** $P4_122$

This group is generated by three translations $T_{1,2,3}$ as given in Eqs. (88), a two-fold screw $S_2$, a two-fold rotation $C_2'$, and a two-fold rotation $C_2''$:

$$S_2: (x,y,z) \to (-x,-y,z+1/2), \tag{412a}$$
$$C_2': (x,y,z) \to (-x,y,-z), \tag{412b}$$
$$C_2'': (x,y,z) \to (-y,-x,-z+1/4). \tag{412c}$$

The $\mathbb{Z}_2$ cohomology ring is given by

$$\mathbb{Z}_2[A_{c'}, A_{c''}, A_{x+y}, B_{xy}]/\langle \mathcal{R}_2, \mathcal{R}_3, \mathcal{R}_4 \rangle \tag{413}$$

where the relations are

$$\mathcal{R}_2: \quad A_{c'}A_{c''}, \quad A_{c''}A_{x+y}, \tag{414a}$$
$$\mathcal{R}_3: \quad A_{c'}A_{x+y}^2 + A_{x+y}^3 + A_{c'}B_{xy}, \quad A_{x+y}(A_{c'}A_{x+y} + A_{x+y}^2 + B_{xy}), \tag{414b}$$
$$\mathcal{R}_4: \quad B_{xy}^2. \tag{414c}$$

We have the following table regarding IWPs and group cohomology at degree 3.

| Wyckoff position | Little group Intl. | Schönflies | Coordinates | LSM anomaly class | Topo. inv. |
|---|---|---|---|---|---|
| 4a | 2 | $C_2$ | $(0,y,0)$, $(0,-y,1/2)$, $(-y,0,1/4)$, $(y,0,3/4)$ | $A_{x+y}(A_{c'}^2 + A_{x+y}^2)$ | $\varphi_2[T_2, C_2']$ |
| 4b | 2 | $C_2$ | $(1/2,y,0)$, $(1/2,-y,1/2)$, $(-y,1/2,1/4)$, $(y,1/2,3/4)$ | $A_{x+y}^2(A_{c'} + A_{x+y})$ | $\varphi_2[T_2, T_1C_2']$ |
| 4c | 2 | $C_2$ | $(x,x,3/8)$, $(-x,-x,7/8)$, $(-x,x,5/8)$, $(x,-x,1/8)$ | $A_{c''}B_{xy}$ | $\varphi_2[T_1T_2^{-1}, C_2'']$ |

**No. 92:** $P4_12_12$

This group is generated by three translations $T_{1,2,3}$ as given in Eqs. (88), a two-fold screw $S_2$, a two-fold screw $S_2'$, and a two-fold rotation $C_2''$:

$$S_2: (x,y,z) \to (-x,-y,z+1/2), \tag{415a}$$
$$S_2': (x,y,z) \to (-x+1/2, y+1/2, -z+1/4), \tag{415b}$$
$$C_2'': (x,y,z) \to (-y,-x,-z+1/2). \tag{415c}$$

The $\mathbb{Z}_2$ cohomology ring is given by

$$\mathbb{Z}_2[A_{c'}, A_{c''}, B_\beta, C_\gamma]/\langle \mathcal{R}_2, \mathcal{R}_3, \mathcal{R}_4, \mathcal{R}_5, \mathcal{R}_6 \rangle \tag{416}$$

where the relations are

$$\mathcal{R}_2: \quad A_{c'}A_{c''}, \quad A_{c'}^2, \tag{417a}$$
$$\mathcal{R}_3: \quad A_{c'}B_\beta, \quad A_{c''}B_\beta, \tag{417b}$$
$$\mathcal{R}_4: \quad A_{c'}C_\gamma, \quad B_\beta^2, \tag{417c}$$
$$\mathcal{R}_5: \quad B_\beta C_\gamma, \tag{417d}$$
$$\mathcal{R}_6: \quad C_\gamma^2. \tag{417e}$$

We have the following table regarding IWPs and group cohomology at degree 3.

| Wyckoff position | Little group Intl. | Schönflies | Coordinates | LSM anomaly class | Topo. inv. |
|---|---|---|---|---|---|
| 4a | 2 | $C_2$ | $(x,x,0)$, $(-x,-x,1/2)$, $(-x+1/2, x+1/2, 1/4)$, $(x+1/2, -x+1/2, 3/4)$ | $C_\gamma$ | $\varphi_2[T_1T_2^{-1}, C_2'']$ |

**No. 93:** $P4_222$

This group is generated by three translations $T_{1,2,3}$ as given in Eqs. (88), a two-fold rotation $C_2$, a two-fold rotation $C_2'$, and a two-fold rotation $C_2''$:

$$C_2: (x,y,z) \to (-x,-y,z), \tag{418a}$$
$$C_2': (x,y,z) \to (-x,y,-z), \tag{418b}$$
$$C_2'': (x,y,z) \to (-y,-x,-z+1/2). \tag{418c}$$

The $\mathbb{Z}_2$ cohomology ring is given by

$$\mathbb{Z}_2[A_{c'}, A_{c''}, A_{x+y}, A_z, B_{xy}]/\langle \mathcal{R}_2, \mathcal{R}_3, \mathcal{R}_4 \rangle \tag{419}$$

where the relations are

$$\mathcal{R}_2: \quad A_{c'} A_{c''}, \quad A_{c''} A_{x+y}, \tag{420a}$$
$$\mathcal{R}_3: \quad A_{c'} A_{x+y}^2 + A_{x+y}^3 + A_{c'} A_{x+y} A_z + A_{x+y} A_z^2 + A_{c'} B_{xy}, \quad A_{x+y}(A_{c'} A_{x+y} + A_{x+y}^2 + A_{c'} A_z + A_z^2 + B_{xy}), \tag{420b}$$
$$\mathcal{R}_4: \quad A_{c'} A_{x+y}^2 A_z + A_{x+y}^3 A_z + A_{c'} A_{x+y} A_z^2 + A_{x+y} A_z^3 + A_{c''} A_z B_{xy} + A_z^2 B_{xy} + B_{xy}^2. \tag{420c}$$

We have the following table regarding IWPs and group cohomology at degree 3.

| Wyckoff position | Little group Intl. | Schönflies | Coordinates | LSM anomaly class | Topo. inv. |
|---|---|---|---|---|---|
| 2a | 222 | $D_2$ | $(0,0,0)$, $(0,0,1/2)$ | $(A_{c'}+A_{x+y})(A_{x+y}+A_z)(A_{c'}+A_{x+y}+A_z)$ | $\varphi_2[C_2, C_2']$ |
| 2b | 222 | $D_2$ | $(1/2,1/2,0)$, $(1/2,1/2,1/2)$ | $A_{x+y}(A_{x+y}+A_z)(A_{c'}+A_{x+y}+A_z)$ | $\varphi_2[T_1 T_2 C_2, T_1 C_2']$ |
| 2c | 222 | $D_2$ | $(0,1/2,0)$, $(1/2,0,1/2)$ | $A_{x+y}(A_{c'}+A_{x+y})(A_{c'}+A_{x+y}+A_z)$ | $\varphi_2[T_2 C_2, C_2']$ |
| 2d | 222 | $D_2$ | $(0,1/2,1/2)$, $(1/2,0,0)$ | $A_{x+y}(A_{c'}+A_{x+y})(A_{x+y}+A_z)$ | $\varphi_2[T_2 C_2, T_3 C_2']$ |
| 2e | 222 | $D_2$ | $(0,0,1/4)$, $(0,0,3/4)$ | $A_{c''}(A_{c''} A_z + A_z^2 + B_{xy})$ | $\varphi_2[C_2, C_2'']$ |
| 2f | 222 | $D_2$ | $(1/2,1/2,1/4)$, $(1/2,1/2,3/4)$ | $A_{c''} B_{xy}$ | $\varphi_2[T_1 T_2 C_2, T_1 T_2 C_2'']$ |

**No. 94:** $P4_2 2_1 2$

This group is generated by three translations $T_{1,2,3}$ as given in Eqs. (88), a two-fold rotation $C_2$, a two-fold screw $S_2'$, and a two-fold rotation $C_2''$:

$$C_2: (x,y,z) \to (-x,-y,z), \tag{421a}$$
$$S_2': (x,y,z) \to (-x+1/2, y+1/2, -z+1/2), \tag{421b}$$
$$C_2'': (x,y,z) \to (-y,-x,-z). \tag{421c}$$

The $\mathbb{Z}_2$ cohomology ring is given by

$$\mathbb{Z}_2[A_{c'}, A_{c''}, A_z, B_\beta, C_\gamma]/\langle \mathcal{R}_2, \mathcal{R}_3, \mathcal{R}_4, \mathcal{R}_5, \mathcal{R}_6 \rangle \tag{422}$$

where the relations are

$$\mathcal{R}_2: \quad A_{c'} A_{c''}, \quad A_{c'}^2, \tag{423a}$$
$$\mathcal{R}_3: \quad A_{c'}(A_z^2 + B_\beta), \quad A_{c'} A_z^2 + A_{c''} B_\beta, \tag{423b}$$
$$\mathcal{R}_4: \quad A_{c'} C_\gamma, \quad B_\beta(A_z^2 + B_\beta), \tag{423c}$$
$$\mathcal{R}_5: \quad B_\beta C_\gamma, \tag{423d}$$
$$\mathcal{R}_6: \quad C_\gamma(A_{c''}^2 A_z + A_{c''} A_z^2 + C_\gamma). \tag{423e}$$

We have the following table regarding IWPs and group cohomology at degree 3.

| Wyckoff position | Little group Intl. | Schönflies | Coordinates | LSM anomaly class | Topo. inv. |
|---|---|---|---|---|---|
| 2a | 222 | $D_2$ | $(0,0,0)$, $(1/2,1/2,1/2)$ | $A_{c''}^2 A_z + A_{c'} A_z^2 + A_{c''} A_z^2 + C_\gamma$ | $\varphi_2[C_2, C_2'']$ |
| 2b | 222 | $D_2$ | $(0,0,1/2)$, $(1/2,1/2,0)$ | $C_\gamma$ | $\varphi_2[C_2, T_3 C_2'']$ |
| 4d | 2 | $C_2$ | $(0,1/2,z)$, $(0,1/2,z+1/2)$, $(1/2,0,-z+1/2)$, $(1/2,0,-z)$ | $A_{c'} A_z^2$ | $\varphi_2[S_2' C_2'', T_1 C_2]$ |

**No. 95:** $P4_322$

This group is generated by three translations $T_{1,2,3}$ as given in Eqs. (88), a two-fold screw $S_2$, a two-fold rotation $C_2'$, and a two-fold rotation $C_2''$:

$$S_2\colon (x,y,z) \to (-x,-y,z+1/2), \tag{424a}$$
$$C_2'\colon (x,y,z) \to (-x,y,-z), \tag{424b}$$
$$C_2''\colon (x,y,z) \to (-y,-x,-z+3/4). \tag{424c}$$

The $\mathbb{Z}_2$ cohomology ring is given by

$$\mathbb{Z}_2[A_{c'}, A_{c''}, A_{x+y}, B_{xy}]/\langle \mathscr{R}_2, \mathscr{R}_3, \mathscr{R}_4 \rangle \tag{425}$$

where the relations are

$$\mathscr{R}_2\colon \quad A_{c'}A_{c''}, \quad A_{c''}A_{x+y}, \tag{426a}$$
$$\mathscr{R}_3\colon \quad A_{c'}A_{x+y}^2 + A_{x+y}^3 + A_{c'}B_{xy}, \quad A_{x+y}(A_{c'}A_{x+y} + A_{x+y}^2 + B_{xy}), \tag{426b}$$
$$\mathscr{R}_4\colon \quad B_{xy}^2. \tag{426c}$$

We have the following table regarding IWPs and group cohomology at degree 3.

| Wyckoff position | Little group Intl. | Schönflies | Coordinates | LSM anomaly class | Topo. inv. |
|---|---|---|---|---|---|
| 4a | 2 | $C_2$ | $(0,y,0)$, $(0,-y,1/2)$, $(-y,0,3/4)$, $(y,0,1/4)$ | $A_{x+y}(A_{c'}^2 + A_{x+y}^2)$ | $\varphi_2[T_2, C_2']$ |
| 4b | 2 | $C_2$ | $(1/2,y,0)$, $(1/2,-y,1/2)$, $(-y,1/2,3/4)$, $(y,1/2,1/4)$ | $A_{x+y}^2(A_{c'} + A_{x+y})$ | $\varphi_2[T_2, T_1 C_2']$ |
| 4c | 2 | $C_2$ | $(x,x,5/8)$, $(-x,-x,1/8)$, $(-x,x,3/8)$, $(x,-x,7/8)$ | $A_{c''}B_{xy}$ | $\varphi_2[T_1 T_2^{-1}, C_2'']$ |

**No. 96:** $P4_32_12$

This group is generated by three translations $T_{1,2,3}$ as given in Eqs. (88), a two-fold screw $S_2$, a two-fold screw $S_2'$, and a two-fold rotation $C_2''$:

$$S_2\colon (x,y,z) \to (-x,-y,z+1/2), \tag{427a}$$
$$S_2'\colon (x,y,z) \to (-x+1/2,y+1/2,-z+3/4), \tag{427b}$$
$$C_2''\colon (x,y,z) \to (-y,-x,-z+1/2). \tag{427c}$$

The $\mathbb{Z}_2$ cohomology ring is given by

$$\mathbb{Z}_2[A_{c'}, A_{c''}, B_\beta, C_\gamma]/\langle \mathscr{R}_2, \mathscr{R}_3, \mathscr{R}_4, \mathscr{R}_5, \mathscr{R}_6 \rangle \tag{428}$$

where the relations are

$$\mathscr{R}_2\colon \quad A_{c'}A_{c''}, \quad A_{c'}^2, \tag{429a}$$
$$\mathscr{R}_3\colon \quad A_{c'}B_\beta, \quad A_{c''}B_\beta, \tag{429b}$$
$$\mathscr{R}_4\colon \quad A_{c'}C_\gamma, \quad B_\beta^2, \tag{429c}$$
$$\mathscr{R}_5\colon \quad B_\beta C_\gamma, \tag{429d}$$
$$\mathscr{R}_6\colon \quad C_\gamma^2. \tag{429e}$$

We have the following table regarding IWPs and group cohomology at degree 3.

| Wyckoff position | Little group Intl. | Schönflies | Coordinates | LSM anomaly class | Topo. inv. |
|---|---|---|---|---|---|
| 4a | 2 | $C_2$ | $(x,x,0)$, $(-x,-x,1/2)$, $(-x+1/2,x+1/2,3/4)$, $(x+1/2,-x+1/2,1/4)$ | $C_\gamma$ | $\varphi_2[T_1 T_2, T_3^{-1} S_2 C_2'']$ |

**No. 97:** $I422$

This group is generated by three translations $T_{1,2,3}$ as given in Eqs. (91), a two-fold rotation $C_2$, a two-fold rotation $C_2'$, and a two-fold rotation $C_2''$:

$$C_2\colon (x,y,z) \to (-x,-y,z), \tag{430a}$$
$$C_2'\colon (x,y,z) \to (-x,y,-z), \tag{430b}$$
$$C_2''\colon (x,y,z) \to (-y,-x,-z). \tag{430c}$$

The $\mathbb{Z}_2$ cohomology ring is given by

$$\mathbb{Z}_2[A_{c'}, A_{c''}, A_{x+y+z}, B_\alpha, B_\beta, B_{z(x+y)}, C_\gamma, C_{xyz}]/\langle \mathcal{R}_2, \mathcal{R}_3, \mathcal{R}_4, \mathcal{R}_5, \mathcal{R}_6 \rangle \tag{431}$$

where the relations are

$$\mathcal{R}_2\colon \quad A_{c'}A_{c''}, \quad A_{c'}A_{x+y+z}, \quad A_{x+y+z}(A_{c''}+A_{x+y+z}), \tag{432a}$$

$$\mathcal{R}_3\colon \quad A_{x+y+z}B_\alpha + A_{c''}B_\beta, \quad A_{x+y+z}(B_\alpha+B_\beta), \quad A_{x+y+z}B_\alpha + A_{c''}B_{z(x+y)}, \quad A_{x+y+z}(B_\alpha+B_{z(x+y)}), \tag{432b}$$

$$\mathcal{R}_4\colon \quad A_{c''}C_\gamma + A_{x+y+z}C_\gamma + A_{c''}C_{xyz}, \quad A_{x+y+z}C_{xyz}, \quad A_{c'}^2 B_\beta + B_\alpha B_\beta + A_{c'}^2 B_{z(x+y)} + B_\alpha B_{z(x+y)} + A_{c'}C_\gamma,$$
$$B_\alpha B_\beta + B_\beta^2 + A_{c'}^2 B_{z(x+y)} + A_{c'}C_\gamma, \quad B_\alpha B_\beta + B_\beta B_{z(x+y)} + A_{c'}C_{xyz},$$
$$A_{c'}^2 B_\beta + B_\alpha B_\beta + A_{c'}^2 B_{z(x+y)} + B_{z(x+y)}^2 + A_{c'}C_\gamma, \tag{432c}$$

$$\mathcal{R}_5\colon \quad A_{c'}^3 B_\beta + A_{c'}B_\alpha B_\beta + B_\beta C_\gamma + B_{z(x+y)}C_\gamma,$$
$$A_{c'}^3 B_\beta + A_{c'}B_\alpha B_\beta + B_\alpha C_\gamma + B_\beta C_\gamma + A_{c'}^2 C_{xyz} + B_\alpha C_{xyz}, \quad (A_{c'}^2 + B_\beta)C_{xyz}, \quad B_{z(x+y)}C_{xyz}, \tag{432d}$$

$$\mathcal{R}_6\colon \quad A_{c'}^4 B_\beta + B_\alpha^2 B_\beta + A_{c''}B_\alpha C_\gamma + C_\gamma^2,$$
$$A_{c'}^4 B_\beta + A_{c'}^2 B_\alpha B_\beta + A_{c'}B_\alpha C_\gamma + A_{c''}B_\alpha C_\gamma + A_{x+y+z}B_\alpha C_\gamma + A_{c'}B_\beta C_\gamma + C_\gamma C_{xyz},$$
$$A_{c'}^4 B_\beta + A_{c'}^2 B_\alpha B_\beta + A_{c'}B_\alpha C_\gamma + A_{c''}B_\alpha C_\gamma + A_{x+y+z}B_\alpha C_\gamma + A_{c'}B_\beta C_\gamma + A_{c'}^3 C_{xyz} + C_{xyz}^2. \tag{432e}$$

We have the following table regarding IWPs and group cohomology at degree 3.

| Wyckoff position | Little group Intl. | Little group Schönflies | Coordinates $(0,0,0)+$ $(1/2,1/2,1/2)+$ | LSM anomaly class | Topo. inv. |
|---|---|---|---|---|---|
| 2a | 422 | $D_4$ | $(0,0,0)$ | $(A_{c'}+A_{c''}+A_{x+y+z})B_\alpha + A_{c'}B_{z(x+y)} + C_{xyz}$ | $\varphi_2[C_2, C_2']$ |
| 2b | 422 | $D_4$ | $(0,0,1/2)$ | $C_{xyz}$ | $\varphi_2[C_2, T_1T_2C_2']$ |
| 4c | 222 | $D_2$ | $(0,1/2,0),\ (1/2,0,0)$ | $A_{c'}B_{z(x+y)}$ | $\varphi_2[T_2T_3C_2, T_2T_3C_2']$ |
| 4d | 222 | $D_2$ | $(0,1/2,1/4),\ (1/2,0,1/4)$ | $A_{x+y+z}B_\alpha$ | $\varphi_2[T_2T_3C_2, T_1T_2T_3C_2'']$ |

**No. 98:** $I4_122$

This group is generated by three translations $T_{1,2,3}$ as given in Eqs. (91), a two-fold rotation $C_2$, a two-fold rotation $C_2'$, and a two-fold rotation $C_2''$:

$$C_2\colon (x,y,z) \to (-x,-y,z), \tag{433a}$$
$$C_2'\colon (x,y,z) \to (-x+1/2,y,-z+3/4), \tag{433b}$$
$$C_2''\colon (x,y,z) \to (-y,-x,-z). \tag{433c}$$

The $\mathbb{Z}_2$ cohomology ring is given by

$$\mathbb{Z}_2[A_{c'}, A_{c''}, A_{x+y+z}, B_{z(x+y)}, C_\gamma]/\langle \mathcal{R}_2, \mathcal{R}_3, \mathcal{R}_4, \mathcal{R}_5, \mathcal{R}_6 \rangle \tag{434}$$

where the relations are

$$\mathcal{R}_2\colon \quad A_{c'}A_{c''}, \quad A_{c'}A_{x+y+z}, \tag{435a}$$
$$\mathcal{R}_3\colon \quad A_{c''}B_{z(x+y)}, \quad A_{x+y+z}B_{z(x+y)}, \tag{435b}$$
$$\mathcal{R}_4\colon \quad A_{c'}C_\gamma, \quad B_{z(x+y)}^2, \tag{435c}$$
$$\mathcal{R}_5\colon \quad B_{z(x+y)}C_\gamma, \tag{435d}$$
$$\mathcal{R}_6\colon \quad C_\gamma(A_{c''}^2 A_{x+y+z} + A_{c''}A_{x+y+z}^2 + C_\gamma). \tag{435e}$$

We have the following table regarding IWPs and group cohomology at degree 3.

| Wyckoff position | Little group Intl. | Little group Schönflies | Coordinates $(0,0,0)+$ $(1/2,1/2,1/2)+$ | LSM anomaly class | Topo. inv. |
|---|---|---|---|---|---|
| 4a | 222 | $D_2$ | $(0,0,0), (0,1/2,1/4)$ | $A_{c''}^2 A_{x+y+z} + A_{c''} A_{x+y+z}^2 + C_\gamma$ | $\varphi_2[C_2, C_2'']$ |
| 4b | 222 | $D_2$ | $(0,0,1/2), (0,1/2,3/4)$ | $C_\gamma$ | $\varphi_2[C_2, T_1 T_2 C_2'']$ |
| 8f | 2 | $C_2$ | $(x,1/4,1/8), (-x+1/2,1/4,5/8),$ $(3/4,x+1/2,3/8), (3/4,-x,7/8)$ | $A_{c'} B_{z(x+y)}$ | $\varphi_2[T_1 T_3, C_2']$ |

### No. 99: $P4mm$

This group is generated by three translations $T_{1,2,3}$ as given in Eqs. (88), a two-fold rotation $C_2$, a mirror $M'$, and a mirror $M$:

$$C_2: (x,y,z) \to (-x,-y,z), \tag{436a}$$

$$M': (x,y,z) \to (x,-y,z), \tag{436b}$$

$$M: (x,y,z) \to (y,x,z). \tag{436c}$$

The $\mathbb{Z}_2$ cohomology ring is given by

$$\mathbb{Z}_2[A_{m'}, A_m, A_{x+y}, A_z, B_\alpha, B_{xy}]/\langle \mathcal{R}_2, \mathcal{R}_3, \mathcal{R}_4 \rangle \tag{437}$$

where the relations are

$$\mathcal{R}_2: \quad A_m A_{m'}, \quad A_m A_{x+y}, \quad A_z^2, \tag{438a}$$

$$\mathcal{R}_3: \quad A_{m'} A_{x+y}^2 + A_{x+y}^3 + A_{x+y} B_\alpha + A_{m'} B_{xy}, \quad A_{x+y}(A_{m'} A_{x+y} + A_{x+y}^2 + B_\alpha + B_{xy}), \tag{438b}$$

$$\mathcal{R}_4: \quad B_{xy}(B_\alpha + B_{xy}). \tag{438c}$$

We have the following table regarding IWPs and group cohomology at degree 3.

| Wyckoff position | Little group Intl. | Little group Schönflies | Coordinates | LSM anomaly class | Topo. inv. |
|---|---|---|---|---|---|
| 1a | 4mm | $C_{4v}$ | $(0,0,z)$ | $A_z(A_{m'} A_{x+y} + A_{x+y}^2 + B_\alpha + B_{xy})$ | $\varphi_2[T_3, C_2]$ |
| 1b | 4mm | $C_{4v}$ | $(1/2,1/2,z)$ | $A_z B_{xy}$ | $\varphi_2[T_3, T_1 T_2 C_2]$ |
| 2c | mm2 | $C_{2v}$ | $(1/2,0,z), (0,1/2,z)$ | $A_{x+y}(A_{m'} + A_{x+y})A_z$ | $\varphi_2[T_3, T_1 C_2]$ |

### No. 100: $P4bm$

This group is generated by three translations $T_{1,2,3}$ as given in Eqs. (88), a two-fold rotation $C_2$, a glide $G'$, and a glide $G$:

$$C_2: (x,y,z) \to (-x,-y,z), \tag{439a}$$

$$G': (x,y,z) \to (x+1/2,-y+1/2,z), \tag{439b}$$

$$G: (x,y,z) \to (y+1/2,x+1/2,z). \tag{439c}$$

The $\mathbb{Z}_2$ cohomology ring is given by

$$\mathbb{Z}_2[A_{m'}, A_m, A_z, B_\alpha, B_\beta, C_\gamma]/\langle \mathcal{R}_2, \mathcal{R}_3, \mathcal{R}_4, \mathcal{R}_5, \mathcal{R}_6 \rangle \tag{440}$$

where the relations are

$$\mathcal{R}_2: \quad A_m A_{m'}, \quad A_{m'}^2, \quad A_z^2, \tag{441a}$$

$$\mathcal{R}_3: \quad A_{m'}(B_\alpha + B_\beta), \quad A_{m'} B_\alpha + A_m B_\beta, \tag{441b}$$

$$\mathcal{R}_4: \quad A_{m'} C_\gamma, \quad B_\beta(B_\alpha + B_\beta), \tag{441c}$$

$$\mathcal{R}_5: \quad B_\beta C_\gamma, \tag{441d}$$

$$\mathcal{R}_6: \quad A_m^6 + A_m^4 B_\alpha + B_\alpha^3 + B_\alpha^2 B_\beta + A_m B_\alpha C_\gamma + C_\gamma^2. \tag{441e}$$

We have the following table regarding IWPs and group cohomology at degree 3.

| Wyckoff position | Little group Intl. | Little group Schönflies | Coordinates | LSM anomaly class | Topo. inv. |
|---|---|---|---|---|---|
| 2a | 4 | $C_4$ | $(0,0,z)$, $(1/2,1/2,z)$ | $A_z B_\beta$ | $\varphi_2[T_3, C_2]$ |
| 2b | $mm2$ | $C_{2v}$ | $(1/2,0,z)$, $(0,1/2,z)$ | $A_z(B_\alpha + B_\beta)$ | $\varphi_2[T_3, T_1 C_2]$ |

## No. 101: $P4_2cm$

This group is generated by three translations $T_{1,2,3}$ as given in Eqs. (88), a two-fold rotation $C_2$, a glide $G'$, and a mirror $M$:

$$C_2\colon (x,y,z) \to (-x,-y,z), \tag{442a}$$
$$G'\colon (x,y,z) \to (x,-y,z+1/2), \tag{442b}$$
$$M\colon (x,y,z) \to (y,x,z). \tag{442c}$$

The $\mathbb{Z}_2$ cohomology ring is given by

$$\mathbb{Z}_2[A_{m'}, A_m, A_{x+y}, B_\alpha, B_\beta, B_{xy}]/\langle \mathcal{R}_2, \mathcal{R}_3, \mathcal{R}_4 \rangle \tag{443}$$

where the relations are

$$\mathcal{R}_2\colon \quad A_m A_{m'}, \quad A_m A_{x+y}, \quad A_{m'}^2, \tag{444a}$$
$$\mathcal{R}_3\colon \quad A_{m'} B_\beta, \quad A_{x+y}(A_{m'}A_{x+y} + B_\beta), \quad A_{m'}A_{x+y}^2 + A_{x+y}^3 + A_{x+y}B_\alpha + A_{m'}B_{xy}, \quad A_{x+y}(A_{m'}A_{x+y} + A_{x+y}^2 + B_\alpha + B_{xy}), \tag{444b}$$
$$\mathcal{R}_4\colon \quad B_\beta^2, \quad B_{xy}(B_\alpha + B_{xy}). \tag{444c}$$

We have the following table regarding IWPs and group cohomology at degree 3.

| Wyckoff position | Little group Intl. | Little group Schönflies | Coordinates | LSM anomaly class | Topo. inv. |
|---|---|---|---|---|---|
| 2a | $mm2$ | $C_{2v}$ | $(0,0,z)$, $(0,0,z+1/2)$ | $A_{x+y}^3 + A_{m'}B_\alpha + A_{x+y}B_\alpha$ | $\varphi_2[G', C_2]$ |
| 2b | $mm2$ | $C_{2v}$ | $(1/2,1/2,z)$, $(1/2,1/2,z+1/2)$ | $A_{x+y}(A_{m'}A_{x+y} + A_{x+y}^2 + B_\alpha)$ | $\varphi_2[T_2 G', T_1 T_2 C_2]$ |
| 4c | 2 | $C_2$ | $(0,1/2,z)$, $(1/2,0,z+1/2)$, $(0,1/2,z+1/2)$, $(1/2,0,z)$ | $A_{m'}A_{x+y}^2$ | $\varphi_2[T_2 G', T_2 C_2]$ |

## No. 102: $P4_2nm$

This group is generated by three translations $T_{1,2,3}$ as given in Eqs. (88), a two-fold rotation $C_2$, a glide $G'$, and a mirror $M$:

$$C_2\colon (x,y,z) \to (-x,-y,z), \tag{445a}$$
$$G'\colon (x,y,z) \to (x+1/2,-y+1/2,z+1/2), \tag{445b}$$
$$M\colon (x,y,z) \to (y,x,z). \tag{445c}$$

The $\mathbb{Z}_2$ cohomology ring is given by

$$\mathbb{Z}_2[A_{m'}, A_m, A_{x+y+z}, B_\alpha, C_\gamma]/\langle \mathcal{R}_2, \mathcal{R}_3, \mathcal{R}_4, \mathcal{R}_5, \mathcal{R}_6 \rangle \tag{446}$$

where the relations are

$$\mathcal{R}_2\colon \quad A_m A_{m'}, \quad A_{m'}^2, \tag{447a}$$
$$\mathcal{R}_3\colon \quad (A_m + A_{m'})A_{x+y+z}^2, \quad A_{m'}(A_{x+y+z}^2 + B_\alpha), \tag{447b}$$
$$\mathcal{R}_4\colon \quad A_{x+y+z}^2(A_{x+y+z}^2 + B_\alpha), \quad A_{m'}C_\gamma, \tag{447c}$$
$$\mathcal{R}_5\colon \quad A_{x+y+z}^2 C_\gamma, \tag{447d}$$
$$\mathcal{R}_6\colon \quad A_m^6 + A_{x+y+z}^6 + A_m^4 B_\alpha + B_\alpha^3 + A_m B_\alpha C_\gamma + C_\gamma^2. \tag{447e}$$

We have the following table regarding IWPs and group cohomology at degree 3.

| Wyckoff position | Little group Intl. | Little group Schönflies | Coordinates | LSM anomaly class | Topo. inv. |
|---|---|---|---|---|---|
| 2a | $mm2$ | $C_{2v}$ | $(0,0,z)$, $(1/2,1/2,z+1/2)$ | $A_{x+y+z}(A_{x+y+z}^2 + B_\alpha)$ | $\varphi_2[T_3, C_2]$ |
| 4b | $2$ | $C_2$ | $(0,1/2,z)$, $(0,1/2,z+1/2)$, $(1/2,0,z+1/2)$, $(1/2,0,z)$ | $A_{m'}A_{x+y+z}^2$ | $\varphi_2[G'M, T_1 C_2]$ |

## No. 103: $P4cc$

This group is generated by three translations $T_{1,2,3}$ as given in Eqs. (88), a two-fold rotation $C_2$, a glide $G'$, and a glide $G$:

$$C_2 \colon (x,y,z) \to (-x,-y,z), \tag{448a}$$
$$G' \colon (x,y,z) \to (x,-y,z+1/2), \tag{448b}$$
$$G \colon (x,y,z) \to (y,x,z+1/2). \tag{448c}$$

The $\mathbb{Z}_2$ cohomology ring is given by

$$\mathbb{Z}_2[A_{m'}, A_m, A_{x+y}, B_\alpha, B_{xy}]/\langle \mathcal{R}_2, \mathcal{R}_3, \mathcal{R}_4\rangle \tag{449}$$

where the relations are

$$\mathcal{R}_2\colon \quad A_m A_{m'}, \quad A_m A_{x+y}, \quad (A_m + A_{m'})^2, \tag{450a}$$
$$\mathcal{R}_3\colon \quad A_{m'}A_{x+y}^2 + A_{x+y}^3 + A_{x+y}B_\alpha + A_{m'}B_{xy}, \quad A_{x+y}(A_{m'}A_{x+y} + A_{x+y}^2 + B_\alpha + B_{xy}), \tag{450b}$$
$$\mathcal{R}_4\colon \quad B_{xy}(B_\alpha + B_{xy}). \tag{450c}$$

We have the following table regarding IWPs and group cohomology at degree 3.

| Wyckoff position | Little group Intl. | Little group Schönflies | Coordinates | LSM anomaly class | Topo. inv. |
|---|---|---|---|---|---|
| 2a | $4$ | $C_4$ | $(0,0,z)$, $(0,0,z+1/2)$ | $A_{x+y}^3 + (A_m + A_{m'} + A_{x+y})B_\alpha + A_m B_{xy}$ | $\varphi_2[G', C_2]$ |
| 2b | $4$ | $C_4$ | $(1/2,1/2,z)$, $(1/2,1/2,z+1/2)$ | $A_{m'}A_{x+y}^2 + A_{x+y}^3 + A_{x+y}B_\alpha + A_m B_{xy}$ | $\varphi_2[T_2 G', T_1 T_2 C_2]$ |
| 4c | $2$ | $C_2$ | $(0,1/2,z)$, $(1/2,0,z)$, $(0,1/2,z+1/2)$, $(1/2,0,z+1/2)$ | $A_{m'}A_{x+y}^2$ | $\varphi_2[G', T_1 C_2]$ |

## No. 104: $P4nc$

This group is generated by three translations $T_{1,2,3}$ as given in Eqs. (88), a two-fold rotation $C_2$, a glide $G'$, and a glide $G$:

$$C_2 \colon (x,y,z) \to (-x,-y,z), \tag{451a}$$
$$G' \colon (x,y,z) \to (x+1/2,-y+1/2,z+1/2), \tag{451b}$$
$$G \colon (x,y,z) \to (y+1/2,x+1/2,z+1/2). \tag{451c}$$

The $\mathbb{Z}_2$ cohomology ring is given by

$$\mathbb{Z}_2[A_{m'}, A_m, B_\alpha, B_{\beta1}, B_{\beta2}, C_{\gamma1}, C_{\gamma2}]/\langle \mathcal{R}_2, \mathcal{R}_3, \mathcal{R}_4, \mathcal{R}_5, \mathcal{R}_6\rangle \tag{452}$$

where the relations are

$$\mathcal{R}_2\colon \quad A_m A_{m'}, \quad A_{m'}^2, \quad A_m^2, \tag{453a}$$
$$\mathcal{R}_3\colon \quad A_{m'}(B_\alpha + B_{\beta1}), \quad A_{m'}B_\alpha + A_m B_{\beta1}, \quad A_{m'}B_{\beta2}, \quad A_m B_\alpha + A_{m'}B_\alpha + A_m B_{\beta2}, \tag{453b}$$
$$\mathcal{R}_4\colon \quad (A_m + A_{m'})C_{\gamma1}, \quad A_{m'}(C_{\gamma1} + C_{\gamma2}), \quad B_\alpha^2 + B_\alpha B_{\beta1} + B_\alpha B_{\beta2} + A_m C_{\gamma2}, \quad B_{\beta1}(B_\alpha + B_{\beta1}), \quad B_{\beta1}B_{\beta2} + A_{m'}C_{\gamma1},$$
$$\qquad B_\alpha^2 + B_\alpha B_{\beta1} + B_{\beta2}^2, \tag{453c}$$
$$\mathcal{R}_5\colon \quad (B_\alpha + B_{\beta1})C_{\gamma1}, \quad B_{\beta2}C_{\gamma1}, \quad A_{m'}B_\alpha^2 + B_\alpha C_{\gamma1} + B_{\beta1}C_{\gamma2}, \quad A_m B_\alpha^2 + B_\alpha C_{\gamma1} + B_\alpha C_{\gamma2} + B_{\beta2}C_{\gamma2}, \tag{453d}$$
$$\mathcal{R}_6\colon \quad C_{\gamma1}^2, \quad C_{\gamma1}(A_{m'}B_\alpha + C_{\gamma2}), \quad B_\alpha^3 + B_\alpha^2 B_{\beta1} + A_{m'}B_\alpha C_{\gamma1} + A_m B_\alpha C_{\gamma2} + C_{\gamma2}^2. \tag{453e}$$

We have the following table regarding IWPs and group cohomology at degree 3.

| Wyckoff position | Little group Intl. | Schönflies | Coordinates | LSM anomaly class | Topo. inv. |
|---|---|---|---|---|---|
| 2a | 4 | $C_4$ | $(0,0,z)$, $(1/2,1/2,z+1/2)$ | $C_{\gamma 1}$ | $\varphi_2[T_3, C_2]$ |
| 4b | 2 | $C_2$ | $(0,1/2,z)$, $(1/2,0,z)$, $(1/2,0,z+1/2)$, $(0,1/2,z+1/2)$ | $(A_m + A_{m'})B_\alpha$ | $\varphi_2[T_1^{-1}G, T_2C_2]$ |

### No. 105: $P4_2mc$

This group is generated by three translations $T_{1,2,3}$ as given in Eqs. (88), a two-fold rotation $C_2$, a mirror $M'$, and a glide $G$:

$$C_2: (x,y,z) \to (-x,-y,z), \tag{454a}$$

$$M': (x,y,z) \to (x,-y,z), \tag{454b}$$

$$G: (x,y,z) \to (y,x,z+1/2). \tag{454c}$$

The $\mathbb{Z}_2$ cohomology ring is given by

$$\mathbb{Z}_2[A_{m'}, A_m, A_{x+y}, B_\alpha, B_\beta, B_{xy}, B_{z(x+y)}]/\langle \mathscr{R}_2, \mathscr{R}_3, \mathscr{R}_4 \rangle \tag{455}$$

where the relations are

$$\mathscr{R}_2: \quad A_m A_{m'}, \quad A_m A_{x+y}, \quad A_m^2, \tag{456a}$$

$$\mathscr{R}_3: \quad A_m B_\beta, \quad A_{m'}A_{x+y}^2 + A_{x+y}^3 + A_{x+y}B_\alpha + A_{m'}B_{xy}, \quad A_{x+y}(A_{m'}A_{x+y} + A_{x+y}^2 + B_\alpha + B_{xy}),$$
$$A_{m'}A_{x+y}^2 + A_{x+y}B_\beta + A_{m'}B_{z(x+y)}, \quad A_m B_{z(x+y)}, \tag{456b}$$

$$\mathscr{R}_4: \quad B_\beta^2, \quad A_{m'}^2 A_{x+y}^2 + A_{m'}A_{x+y}B_\alpha + A_{x+y}^2 B_\beta + B_\beta B_{xy} + A_{x+y}^2 B_{z(x+y)} + B_\alpha B_{z(x+y)}, \quad B_\beta(A_{x+y}^2 + B_{z(x+y)}),$$
$$B_{xy}(B_\alpha + B_{xy}), \quad A_{m'}A_{x+y}^3 + A_{x+y}^2 B_\beta + A_{x+y}^2 B_{z(x+y)} + B_\alpha B_{z(x+y)} + B_{xy}B_{z(x+y)}, \quad (A_{x+y}^2 + B_{z(x+y)})^2. \tag{456c}$$

We have the following table regarding IWPs and group cohomology at degree 3.

| Wyckoff position | Little group Intl. | Schönflies | Coordinates | LSM anomaly class | Topo. inv. |
|---|---|---|---|---|---|
| 2a | $mm2$ | $C_{2v}$ | $(0,0,z)$, $(0,0,z+1/2)$ | $A_m(B_\alpha + B_{xy})$ | $\varphi_2[G, C_2]$ |
| 2b | $mm2$ | $C_{2v}$ | $(1/2,1/2,z)$, $(1/2,1/2,z+1/2)$ | $A_m B_{xy}$ | $\varphi_2[G, T_1T_2C_2]$ |
| 2c | $mm2$ | $C_{2v}$ | $(0,1/2,z)$, $(1/2,0,z+1/2)$ | $A_{x+y}(A_{x+y}^2 + B_\beta + B_{z(x+y)})$ | $\varphi_2[T_3, T_1C_2]$ |

### No. 106: $P4_2bc$

This group is generated by three translations $T_{1,2,3}$ as given in Eqs. (88), a two-fold rotation $C_2$, a glide $G'$, and a glide $G$:

$$C_2: (x,y,z) \to (-x,-y,z), \tag{457a}$$

$$G': (x,y,z) \to (x+1/2, -y+1/2, z), \tag{457b}$$

$$G: (x,y,z) \to (y+1/2, x+1/2, z+1/2). \tag{457c}$$

The $\mathbb{Z}_2$ cohomology ring is given by

$$\mathbb{Z}_2[A_{m'}, A_m, B_\alpha, B_{\beta 1}, B_{\beta 2}, C_{\gamma 1}, C_{\gamma 2}]/\langle \mathscr{R}_2, \mathscr{R}_3, \mathscr{R}_4, \mathscr{R}_5, \mathscr{R}_6 \rangle \tag{458}$$

where the relations are

$$\mathscr{R}_2: \quad A_m A_{m'}, \quad A_{m'}^2, \quad A_m^2, \tag{459a}$$

$$\mathscr{R}_3: \quad A_{m'}(B_\alpha + B_{\beta 1}), \quad A_{m'}B_\alpha + A_m B_{\beta 1}, \quad A_{m'}B_{\beta 2}, \quad A_m B_{\beta 2}, \tag{459b}$$

$$\mathscr{R}_4: \quad A_{m'}C_{\gamma 1}, \quad A_m C_{\gamma 1} + A_m C_{\gamma 2} + A_{m'}C_{\gamma 2}, \quad B_\alpha B_{\beta 2} + A_m C_{\gamma 1} + A_{m'}C_{\gamma 2}, \quad B_{\beta 1}(B_\alpha + B_{\beta 1}), \quad B_{\beta 1}B_{\beta 2} + A_{m'}C_{\gamma 2}, \quad B_{\beta 2}^2, \tag{459c}$$

$$\mathscr{R}_5: \quad A_{m'}B_\alpha^2 + B_{\beta 1}C_{\gamma 1}, \quad A_m B_\alpha^2 + A_{m'}B_\alpha^2 + B_{\beta 2}C_{\gamma 1}, \quad A_{m'}B_\alpha^2 + B_\alpha C_{\gamma 1} + B_\alpha C_{\gamma 2} + B_{\beta 1}C_{\gamma 2}, \quad A_m B_\alpha^2 + B_{\beta 2}C_{\gamma 2}, \tag{459d}$$

$$\mathscr{R}_6: \quad B_\alpha^3 + B_\alpha^2 B_{\beta 1} + A_m B_\alpha C_{\gamma 1} + C_{\gamma 1}^2, \quad B_\alpha^3 + B_\alpha^2 B_{\beta 1} + A_m B_\alpha C_{\gamma 1} + A_{m'}B_\alpha C_{\gamma 2} + C_{\gamma 1}C_{\gamma 2}, \quad B_\alpha^3 + A_m B_\alpha C_{\gamma 1} + C_{\gamma 2}^2. \tag{459e}$$

We have the following table regarding IWPs and group cohomology at degree 3.

| Wyckoff position | Little group Intl. | Little group Schönflies | Coordinates | LSM anomaly class | Topo. inv. |
|---|---|---|---|---|---|
| 4a | 2 | $C_2$ | $(0,0,z)$, $(0,0,z+1/2)$, $(1/2,1/2,z)$, $(1/2,1/2,z+1/2)$ | $A_{m'}B_\alpha$ | $\varphi_2[T_1^{-1}G'G, C_2]$ |
| 4b | 2 | $C_2$ | $(0,1/2,z)$, $(1/2,0,z+1/2)$, $(1/2,0,z)$, $(0,1/2,z+1/2)$ | $(A_m + A_{m'})B_\alpha$ | $\varphi_2[T_1T_2C_2G, T_1C_2]$ |

### No. 107: $I4mm$

This group is generated by three translations $T_{1,2,3}$ as given in Eqs. (91), a two-fold rotation $C_2$, a mirror $M'$, and a mirror $M$:

$$C_2: (x,y,z) \rightarrow (-x,-y,z), \tag{460a}$$
$$M': (x,y,z) \rightarrow (x,-y,z), \tag{460b}$$
$$M: (x,y,z) \rightarrow (y,x,z). \tag{460c}$$

The $\mathbb{Z}_2$ cohomology ring is given by

$$\mathbb{Z}_2[A_{m'}, A_m, A_{x+y+z}, B_\alpha, B_\beta, B_{z(x+y)}, C_{xyz}]/\langle \mathcal{R}_2, \mathcal{R}_3, \mathcal{R}_4, \mathcal{R}_5, \mathcal{R}_6\rangle \tag{461}$$

where the relations are

$$\mathcal{R}_2: \quad A_m A_{m'}, \quad A_{m'}A_{x+y+z}, \quad A_{x+y+z}^2, \tag{462a}$$
$$\mathcal{R}_3: \quad A_{x+y+z}B_\alpha + A_m B_\beta, \quad A_{x+y+z}(B_\alpha + B_\beta), \quad A_{x+y+z}B_\alpha + A_m B_{z(x+y)}, \quad A_{x+y+z}(B_\alpha + B_{z(x+y)}), \tag{462b}$$
$$\mathcal{R}_4: \quad A_{x+y+z}C_{xyz}, \quad B_\beta^2 + B_\alpha B_{z(x+y)}, \quad B_\alpha B_\beta + B_\beta B_{z(x+y)} + A_{m'}C_{xyz}, \quad B_{z(x+y)}(B_\alpha + B_{z(x+y)}), \tag{462c}$$
$$\mathcal{R}_5: \quad B_\beta C_{xyz}, \quad B_{z(x+y)}C_{xyz}, \tag{462d}$$
$$\mathcal{R}_6: \quad C_{xyz}^2. \tag{462e}$$

We have the following table regarding IWPs and group cohomology at degree 3.

| Wyckoff position | Little group Intl. | Little group Schönflies | Coordinates $(0,0,0)+$ $(1/2,1/2,1/2)+$ | LSM anomaly class | Topo. inv. |
|---|---|---|---|---|---|
| 2a | $4mm$ | $C_{4v}$ | $(0,0,z)$ | $C_{xyz}$ | $\varphi_2[T_1T_2, C_2]$ |
| 4b | $mm2$ | $C_{2v}$ | $(0,1/2,z)$, $(1/2,0,z)$ | $A_{x+y+z}B_\alpha$ | $\varphi_2[T_2M, T_2T_3C_2]$ |

### No. 108: $I4cm$

This group is generated by three translations $T_{1,2,3}$ as given in Eqs. (91), a two-fold rotation $C_2$, a glide $G'$, and a glide $G$:

$$C_2: (x,y,z) \rightarrow (-x,-y,z), \tag{463a}$$
$$G': (x,y,z) \rightarrow (x,-y,z+1/2), \tag{463b}$$
$$G: (x,y,z) \rightarrow (y,x,z+1/2). \tag{463c}$$

The $\mathbb{Z}_2$ cohomology ring is given by

$$\mathbb{Z}_2[A_{m'}, A_m, A_{x+y+z}, B_\alpha, B_{z(x+y)}, D_\delta]/\langle \mathcal{R}_2, \mathcal{R}_3, \mathcal{R}_4, \mathcal{R}_5, \mathcal{R}_6\rangle \tag{464}$$

where the relations are

$$\mathcal{R}_2: \quad A_m A_{m'}, \quad A_{m'}A_{x+y+z}, \quad A_m^2 + A_{m'}^2 + A_{x+y+z}^2, \tag{465a}$$
$$\mathcal{R}_3: \quad A_m^2 A_{x+y+z} + A_m A_{x+y+z}^2 + A_{x+y+z}B_\alpha + A_m B_{z(x+y)}, \quad A_{x+y+z}(A_m^2 + A_m A_{x+y+z} + B_\alpha + B_{z(x+y)}), \tag{465b}$$
$$\mathcal{R}_4: \quad B_{z(x+y)}(B_\alpha + B_{z(x+y)}), \tag{465c}$$
$$\mathcal{R}_5: \quad A_{m'}D_\delta, \quad (A_m + A_{x+y+z})D_\delta, \tag{465d}$$
$$\mathcal{R}_6: \quad (B_\alpha + B_{z(x+y)})D_\delta, \tag{465e}$$
$$\mathcal{R}_8: \quad A_m^4 B_\alpha^2 + D_\delta^2. \tag{465f}$$

We have the following table regarding IWPs and group cohomology at degree 3.

| Wyckoff position | Little group Intl. | Schönflies | Coordinates $(0,0,0)+\ (1/2,1/2,1/2)+$ | LSM anomaly class | Topo. inv. |
|---|---|---|---|---|---|
| 4a | 4 | $C_4$ | $(0,0,z),\ (0,0,z+1/2)$ | $(A_m + A_{m'} + A_{x+y+z})B_\alpha + A_{m'}B_{z(x+y)}$ | $\varphi_2[G', C_2]$ |
| 4b | $mm2$ | $C_{2v}$ | $(1/2,0,z),\ (0,1/2,z)$ | $A_{m'}B_{z(x+y)}$ | $\varphi_2[G', T_2T_3C_2]$ |

### No. 109: $I4_1md$

This group is generated by three translations $T_{1,2,3}$ as given in Eqs. (91), a two-fold rotation $C_2$, a mirror $M'$, and a glide $G$:

$$C_2\colon (x,y,z) \to (-x,-y,z), \tag{466a}$$
$$M'\colon (x,y,z) \to (x,-y,z), \tag{466b}$$
$$G\colon (x,y,z) \to (y+1/2, x, z+3/4). \tag{466c}$$

The $\mathbb{Z}_2$ cohomology ring is given by

$$\mathbb{Z}_2[A_{m'}, A_m, B_\alpha, B_{z(x+y)}, C_\beta, C_\gamma, D_\delta]/\langle \mathscr{R}_2, \mathscr{R}_3, \mathscr{R}_4, \mathscr{R}_5, \mathscr{R}_6 \rangle \tag{467}$$

where the relations are

$$\mathscr{R}_2\colon \quad A_m A_{m'}, \quad A_m^2, \tag{468a}$$
$$\mathscr{R}_3\colon \quad A_m B_\alpha, \quad A_m B_{z(x+y)}, \tag{468b}$$
$$\mathscr{R}_4\colon \quad A_m C_\beta, \quad A_m C_\gamma, \quad B_\alpha B_{z(x+y)} + A_{m'}C_\gamma, \quad B_{z(x+y)}^2, \tag{468c}$$
$$\mathscr{R}_5\colon \quad A_m D_\delta, \quad A_{m'}^2 C_\beta + B_{z(x+y)}C_\beta + A_{m'}^2 C_\gamma + A_{m'}D_\delta, \quad B_{z(x+y)}C_\gamma, \tag{468d}$$
$$\mathscr{R}_6\colon \quad A_{m'}^3 C_\beta + A_{m'}^3 C_\gamma + A_{m'}^2 D_\delta + B_{z(x+y)}D_\delta, \quad B_\alpha^3 + A_{m'}B_\alpha C_\beta + C_\beta^2 + A_{m'}^3 C_\gamma,$$
$$\qquad A_{m'}B_\alpha C_\beta + A_{m'}B_\alpha C_\gamma + C_\beta C_\gamma + B_\alpha D_\delta, \quad C_\gamma^2, \tag{468e}$$
$$\mathscr{R}_7\colon \quad A_{m'}B_\alpha^3 + A_{m'}^2 B_\alpha C_\beta + A_{m'}^4 C_\gamma + B_\alpha^2 C_\gamma + C_\beta D_\delta, \quad A_{m'}^2 B_\alpha C_\beta + A_{m'}^2 B_\alpha C_\gamma + A_{m'}B_\alpha D_\delta + C_\gamma D_\delta, \tag{468f}$$
$$\mathscr{R}_8\colon \quad A_{m'}^2 B_\alpha^3 + A_{m'}^3 B_\alpha C_\beta + A_{m'}^5 C_\gamma + D_\delta^2. \tag{468g}$$

We have the following table regarding IWPs and group cohomology at degree 3.

| Wyckoff position | Little group Intl. | Schönflies | Coordinates $(0,0,0)+\ (1/2,1/2,1/2)+$ | LSM anomaly class | Topo. inv. |
|---|---|---|---|---|---|
| 4a | $mm2$ | $C_{2v}$ | $(0,0,z),\ (0,1/2,z+1/4)$ | $C_\gamma$ | $\varphi_2[T_1T_2, C_2]$ |

### No. 110: $I4_1cd$

This group is generated by three translations $T_{1,2,3}$ as given in Eqs. (91), a two-fold rotation $C_2$, a glide $G'$, and a glide $G$:

$$C_2\colon (x,y,z) \to (-x,-y,z), \tag{469a}$$
$$G'\colon (x,y,z) \to (x,-y,z+1/2), \tag{469b}$$
$$G\colon (x,y,z) \to (y+1/2, x, z+1/4). \tag{469c}$$

The $\mathbb{Z}_2$ cohomology ring is given by

$$\mathbb{Z}_2[A_{m'}, A_m, B_\alpha, C_\gamma]/\langle \mathscr{R}_2, \mathscr{R}_3, \mathscr{R}_4, \mathscr{R}_5, \mathscr{R}_6 \rangle \tag{470}$$

where the relations are

$$\mathscr{R}_2\colon \quad A_m A_{m'}, \quad A_m^2, \tag{471a}$$
$$\mathscr{R}_3\colon \quad A_{m'}^3, \quad A_m B_\alpha, \tag{471b}$$
$$\mathscr{R}_4\colon \quad A_{m'}^2 B_\alpha, \quad A_m C_\gamma, \tag{471c}$$
$$\mathscr{R}_5\colon \quad A_{m'}^2 C_\gamma, \tag{471d}$$
$$\mathscr{R}_6\colon \quad B_\alpha^3 + A_{m'}B_\alpha C_\gamma + C_\gamma^2. \tag{471e}$$

We have the following table regarding IWPs and group cohomology at degree 3.

| Wyckoff position | Little group Intl. | Little group Schönflies | Coordinates $(0,0,0)+ \ (1/2,1/2,1/2)+$ | LSM anomaly class | Topo. inv. |
|---|---|---|---|---|---|
| 8a | 2 | $C_2$ | $(0,0,z), \ (0,1/2,z+1/4),$ $(0,0,z+1/2), \ (0,1/2,z+3/4)$ | $A_{m'}B_\alpha$ | $\varphi_2[G',C_2]$ |

## No. 111: $P\bar{4}2m$

This group is generated by three translations $T_{1,2,3}$ as given in Eqs. (88), a two-fold rotation $C_2$, a two-fold rotation $C_2'$, and a mirror $M$:

$$C_2\colon (x,y,z) \to (-x,-y,z), \tag{472a}$$
$$C_2'\colon (x,y,z) \to (-x,y,-z), \tag{472b}$$
$$M\colon (x,y,z) \to (y,x,z). \tag{472c}$$

The $\mathbb{Z}_2$ cohomology ring is given by

$$\mathbb{Z}_2[A_{c'}, A_m, A_{x+y}, A_z, B_\alpha, B_{xy}]/\langle \mathcal{R}_2, \mathcal{R}_3, \mathcal{R}_4\rangle \tag{473}$$

where the relations are

$$\mathcal{R}_2\colon \quad A_{c'}A_m, \quad A_m A_{x+y}, \quad A_z(A_{c'} + A_z), \tag{474a}$$
$$\mathcal{R}_3\colon \quad A_{c'}A_{x+y}^2 + A_{x+y}^3 + A_{x+y}B_\alpha + A_{c'}B_{xy}, \quad A_{x+y}(A_{c'}A_{x+y} + A_{x+y}^2 + B_\alpha + B_{xy}), \tag{474b}$$
$$\mathcal{R}_4\colon \quad B_{xy}(B_\alpha + B_{xy}). \tag{474c}$$

We have the following table regarding IWPs and group cohomology at degree 3.

| Wyckoff position | Little group Intl. | Little group Schönflies | Coordinates | LSM anomaly class | Topo. inv. |
|---|---|---|---|---|---|
| 1a | $\bar{4}2m$ | $D_{2d}$ | $(0,0,0)$ | $(A_{c'} + A_z)(A_{c'}A_{x+y} + A_{x+y}^2 + B_\alpha + B_{xy})$ | $\varphi_2[C_2, C_2']$ |
| 1b | $\bar{4}2m$ | $D_{2d}$ | $(1/2,1/2,1/2)$ | $A_z B_{xy}$ | $\varphi_2[T_1T_2C_2, T_1T_3C_2']$ |
| 1c | $\bar{4}2m$ | $D_{2d}$ | $(0,0,1/2)$ | $A_z(A_{c'}A_{x+y} + A_{x+y}^2 + B_\alpha + B_{xy})$ | $\varphi_2[C_2, T_3C_2']$ |
| 1d | $\bar{4}2m$ | $D_{2d}$ | $(1/2,1/2,0)$ | $(A_{c'} + A_z)B_{xy}$ | $\varphi_2[T_1T_2C_2, T_1C_2']$ |
| 2e | 222 | $D_2$ | $(1/2,0,0), (0,1/2,0)$ | $A_{x+y}(A_{c'} + A_{x+y})(A_{c'} + A_z)$ | $\varphi_2[T_1C_2, T_1C_2']$ |
| 2f | 222 | $D_2$ | $(1/2,0,1/2), (0,1/2,1/2)$ | $A_{x+y}(A_{c'} + A_{x+y})A_z$ | $\varphi_2[T_1C_2, T_1T_3C_2']$ |

## No. 112: $P\bar{4}2c$

This group is generated by three translations $T_{1,2,3}$ as given in Eqs. (88), a two-fold rotation $C_2$, a two-fold rotation $C_2'$, and a glide $G$:

$$C_2\colon (x,y,z) \to (-x,-y,z), \tag{475a}$$
$$C_2'\colon (x,y,z) \to (-x,y,-z), \tag{475b}$$
$$G\colon (x,y,z) \to (y,x,z+1/2). \tag{475c}$$

The $\mathbb{Z}_2$ cohomology ring is given by

$$\mathbb{Z}_2[A_{c'}, A_m, A_{x+y}, B_\alpha, B_{xy}, B_\beta, B_{z(x+y)}]/\langle \mathcal{R}_2, \mathcal{R}_3, \mathcal{R}_4\rangle \tag{476}$$

where the relations are

$$\mathcal{R}_2\colon \quad A_{c'}A_m, \quad A_m A_{x+y}, \quad A_m^2, \tag{477a}$$

$$\mathcal{R}_3\colon \quad A_{c'}A_{x+y}^2 + A_{x+y}^3 + A_{x+y}B_\alpha + A_{c'}B_{xy}, \quad A_{x+y}(A_{c'}A_{x+y} + A_{x+y}^2 + B_\alpha + B_{xy}),$$
$$\quad A_m B_\beta, \quad A_{x+y}B_\beta + A_{c'}B_{z(x+y)}, \quad A_m B_{z(x+y)}, \tag{477b}$$

$$\mathcal{R}_4\colon \quad B_{xy}(B_\alpha + B_{xy}), \quad A_{x+y}^2 B_\beta + B_\beta B_{xy} + A_{x+y}^2 B_{z(x+y)} + B_\alpha B_{z(x+y)},$$
$$\quad A_{x+y}^2 B_\beta + A_{x+y}^2 B_{z(x+y)} + B_\alpha B_{z(x+y)} + B_{xy}B_{z(x+y)}, \quad A_{c'}^2 B_\alpha + A_{c'}^2 B_\beta + B_\beta^2,$$
$$\quad A_{c'}A_{x+y}B_\alpha + A_{c'}A_{x+y}B_\beta + B_\beta B_{z(x+y)}, \quad A_{c'}^2 A_{x+y}^2 + A_{x+y}^4 + A_{c'}A_{x+y}B_\alpha + A_{x+y}^2 B_\beta + B_{z(x+y)}^2. \tag{477c}$$

We have the following table regarding IWPs and group cohomology at degree 3.

| Wyckoff position | Little group Intl. | Schönflies | Coordinates | LSM anomaly class | Topo. inv. |
|---|---|---|---|---|---|
| 2a | 222 | $D_2$ | $(0,0,1/4)$, $(0,0,3/4)$ | $(A_{c'} + A_m)(A_{c'}A_{x+y} + A_{x+y}^2 + B_\alpha + B_{xy})$ | $\varphi_2[C_2, C_2']$ |
| 2b | 222 | $D_2$ | $(1/2,0,1/4)$, $(0,1/2,3/4)$ | $(A_{c'} + A_{x+y})(A_{x+y}^2 + B_{z(x+y)})$ | $\varphi_2[T_1C_2, T_1C_2']$ |
| 2c | 222 | $D_2$ | $(1/2,1/2,1/4)$, $(1/2,1/2,3/4)$ | $(A_{c'} + A_m)B_{xy}$ | $\varphi_2[T_1T_2C_2, T_1C_2']$ |
| 2d | 222 | $D_2$ | $(0,1/2,1/4)$, $(1/2,0,3/4)$ | $(A_{c'} + A_{x+y})(A_{c'}A_{x+y} + A_{x+y}^2 + B_{z(x+y)})$ | $\varphi_2[T_2C_2, C_2']$ |
| 2e | $\bar{4}$ | $S_4$ | $(0,0,0)$, $(0,0,1/2)$ | $A_m(B_\alpha + B_{xy})$ | $\varphi_2[C_2'G, C_2]$ |
| 2f | $\bar{4}$ | $S_4$ | $(1/2,1/2,0)$, $(1/2,1/2,1/2)$ | $A_mB_{xy}$ | $\varphi_2[T_1C_2'G, T_1T_2C_2]$ |

## No. 113: $P\bar{4}2_1m$

This group is generated by three translations $T_{1,2,3}$ as given in Eqs. (88), a two-fold rotation $C_2$, a two-fold screw $S_2'$, and a glide $G$:

$$C_2\colon (x,y,z) \to (-x,-y,z), \tag{478a}$$
$$S_2'\colon (x,y,z) \to (-x+1/2, y+1/2, -z), \tag{478b}$$
$$G\colon (x,y,z) \to (y+1/2, x+1/2, z). \tag{478c}$$

The $\mathbb{Z}_2$ cohomology ring is given by

$$\mathbb{Z}_2[A_{c'}, A_m, A_z, B_\alpha, B_\beta, C_\gamma]/\langle \mathscr{R}_2, \mathscr{R}_3, \mathscr{R}_4, \mathscr{R}_5, \mathscr{R}_6 \rangle \tag{479}$$

where the relations are

$$\mathscr{R}_2\colon \quad A_{c'}A_m, \quad A_{c'}^2, \quad A_z(A_{c'} + A_z), \tag{480a}$$
$$\mathscr{R}_3\colon \quad A_{c'}(B_\alpha + B_\beta), \quad A_{c'}B_\alpha + A_mB_\beta, \tag{480b}$$
$$\mathscr{R}_4\colon \quad A_{c'}C_\gamma, \quad B_\beta(B_\alpha + B_\beta), \tag{480c}$$
$$\mathscr{R}_5\colon \quad A_{c'}B_\alpha^2 + B_\beta C_\gamma, \tag{480d}$$
$$\mathscr{R}_6\colon \quad B_\alpha^3 + B_\alpha^2 B_\beta + A_mB_\alpha C_\gamma + C_\gamma^2. \tag{480e}$$

We have the following table regarding IWPs and group cohomology at degree 3.

| Wyckoff position | Little group Intl. | Schönflies | Coordinates | LSM anomaly class | Topo. inv. |
|---|---|---|---|---|---|
| 2a | $\bar{4}$ | $S_4$ | $(0,0,0)$, $(1/2,1/2,0)$ | $A_{c'}B_\alpha + A_zB_\beta$ | $\varphi_2[T_2^{-1}S_2'G, C_2]$ |
| 2b | $\bar{4}$ | $S_4$ | $(0,0,1/2)$, $(1/2,1/2,1/2)$ | $A_zB_\beta$ | $\varphi_2[T_2^{-1}T_3S_2'G, C_2]$ |
| 2c | $mm2$ | $C_{2v}$ | $(0,1/2,z)$, $(1/2,0,-z)$ | $A_z(B_\alpha + B_\beta)$ | $\varphi_2[T_3, T_1C_2]$ |

## No. 114: $P\bar{4}2_1c$

This group is generated by three translations $T_{1,2,3}$ as given in Eqs. (88), a two-fold rotation $C_2$, a two-fold screw $S_2'$, and a glide $G$:

$$C_2\colon (x,y,z) \to (-x,-y,z), \tag{481a}$$
$$S_2'\colon (x,y,z) \to (-x+1/2, y+1/2, -z+1/2), \tag{481b}$$
$$G\colon (x,y,z) \to (y+1/2, x+1/2, z+1/2). \tag{481c}$$

The $\mathbb{Z}_2$ cohomology ring is given by

$$\mathbb{Z}_2[A_{c'}, A_m, B_\alpha, B_{\beta 1}, B_{\beta 2}, C_{\gamma 1}, C_{\gamma 2}]/\langle \mathscr{R}_2, \mathscr{R}_3, \mathscr{R}_4, \mathscr{R}_5, \mathscr{R}_6 \rangle \tag{482}$$

where the relations are

$$\mathscr{R}_2: \quad A_{c'}A_m, \quad A_{c'}^2, \quad A_m^2, \tag{483a}$$

$$\mathscr{R}_3: \quad A_{c'}(B_\alpha + B_{\beta 1}), \quad A_{c'}B_\alpha + A_m B_{\beta 1}, \quad A_{c'}B_{\beta 2}, \quad A_{c'}B_\alpha + A_m B_\alpha + A_m B_{\beta 2}, \tag{483b}$$

$$\mathscr{R}_4: \quad (A_{c'} + A_m)C_{\gamma 1}, \quad A_{c'}(C_{\gamma 1} + C_{\gamma 2}), \quad B_\alpha^2 + B_\alpha B_{\beta 1} + B_\alpha B_{\beta 2} + A_m C_{\gamma 2}, \quad B_{\beta 1}(B_\alpha + B_{\beta 1}), \quad B_{\beta 1}B_{\beta 2} + A_{c'}C_{\gamma 1},$$
$$B_\alpha^2 + B_\alpha B_{\beta 1} + B_{\beta 2}^2, \tag{483c}$$

$$\mathscr{R}_5: \quad (B_\alpha + B_{\beta 1})C_{\gamma 1}, \quad B_{\beta 2}C_{\gamma 1}, \quad B_\alpha C_{\gamma 1} + B_{\beta 1}C_{\gamma 2}, \quad A_{c'}B_\alpha^2 + A_m B_\alpha^2 + B_\alpha C_{\gamma 1} + B_\alpha C_{\gamma 2} + B_{\beta 2}C_{\gamma 2}, \tag{483d}$$

$$\mathscr{R}_6: \quad C_{\gamma 1}(A_{c'}B_\alpha + C_{\gamma 1}), \quad C_{\gamma 1}(A_{c'}B_\alpha + C_{\gamma 2}), \quad B_\alpha^3 + B_\alpha^2 B_{\beta 1} + A_m B_\alpha C_{\gamma 2} + C_{\gamma 2}^2. \tag{483e}$$

We have the following table regarding IWPs and group cohomology at degree 3.

| Wyckoff position | Little group Intl. | Little group Schönflies | Coordinates | LSM anomaly class | Topo. inv. |
|---|---|---|---|---|---|
| 2a | $\bar{4}$ | $S_4$ | $(0,0,0)$, $(1/2,1/2,1/2)$ | $A_{c'}B_\alpha + C_{\gamma 1}$ | $\varphi_2[T_2^{-1}S_2'G, C_2]$ |
| 2b | $\bar{4}$ | $S_4$ | $(0,0,1/2)$, $(1/2,1/2,0)$ | $C_{\gamma 1}$ | $\varphi_2[T_2^{-1}T_3 S_2'G, C_2]$ |
| 4d | 2 | $C_2$ | $(0,1/2,z)$, $(1/2,0,-z)$, $(1/2,0,-z+1/2)$, $(0,1/2,z+1/2)$ | $(A_{c'} + A_m)B_\alpha$ | $\varphi_2[T_2^{-1}G, T_1 C_2]$ |

### No. 115: $P\bar{4}m2$

This group is generated by three translations $T_{1,2,3}$ as given in Eqs. (88), a two-fold rotation $C_2$, a two-fold rotation $C_2'$, and a mirror $M$:

$$C_2: (x,y,z) \rightarrow (-x,-y,z), \tag{484a}$$

$$C_2': (x,y,z) \rightarrow (-y,-x,-z), \tag{484b}$$

$$M: (x,y,z) \rightarrow (x,-y,z). \tag{484c}$$

The $\mathbb{Z}_2$ cohomology ring is given by

$$\mathbb{Z}_2[A_{c'}, A_m, A_{x+y}, A_z, B_\alpha, B_{xy}]/\langle \mathscr{R}_2, \mathscr{R}_3, \mathscr{R}_4 \rangle \tag{485}$$

where the relations are

$$\mathscr{R}_2: \quad A_{c'}A_m, \quad A_{c'}A_{x+y}, \quad A_z(A_{c'} + A_z), \tag{486a}$$

$$\mathscr{R}_3: \quad A_m A_{x+y}^2 + A_{x+y}^3 + A_{x+y}B_\alpha + A_m B_{xy}, \quad A_{x+y}(A_m A_{x+y} + A_{x+y}^2 + B_\alpha + B_{xy}), \tag{486b}$$

$$\mathscr{R}_4: \quad B_{xy}(B_\alpha + B_{xy}). \tag{486c}$$

We have the following table regarding IWPs and group cohomology at degree 3.

| Wyckoff position | Little group Intl. | Little group Schönflies | Coordinates | LSM anomaly class | Topo. inv. |
|---|---|---|---|---|---|
| 1a | $\bar{4}m2$ | $D_{2d}$ | $(0,0,0)$ | $A_m A_{x+y}A_z + A_{x+y}^2 A_z + (A_{c'} + A_z)(B_\alpha + B_{xy})$ | $\varphi_2[C_2, C_2']$ |
| 1b | $\bar{4}m2$ | $D_{2d}$ | $(1/2,1/2,0)$ | $(A_{c'} + A_z)B_{xy}$ | $\varphi_2[T_1 T_2 C_2, T_1 T_2 C_2']$ |
| 1c | $\bar{4}m2$ | $D_{2d}$ | $(1/2,1/2,1/2)$ | $A_z B_{xy}$ | $\varphi_2[T_1 T_2 C_2, T_1 T_2 T_3 C_2']$ |
| 1d | $\bar{4}m2$ | $D_{2d}$ | $(0,0,1/2)$ | $A_z(A_m A_{x+y} + A_{x+y}^2 + B_\alpha + B_{xy})$ | $\varphi_2[C_2, T_3 C_2']$ |
| 2g | $mm2$ | $C_{2v}$ | $(0,1/2,z)$, $(1/2,0,-z)$ | $A_{x+y}(A_m + A_{x+y})A_z$ | $\varphi_2[T_3, T_1 C_2]$ |

### No. 116: $P\bar{4}c2$

This group is generated by three translations $T_{1,2,3}$ as given in Eqs. (88), a two-fold rotation $C_2$, a two-fold rotation $C_2'$, and a glide $G$:

$$C_2: (x,y,z) \rightarrow (-x,-y,z), \tag{487a}$$

$$C_2': (x,y,z) \rightarrow (-y,-x,-z+1/2), \tag{487b}$$

$$G: (x,y,z) \rightarrow (x,-y,z+1/2). \tag{487c}$$

The $\mathbb{Z}_2$ cohomology ring is given by

$$\mathbb{Z}_2[A_{c'}, A_m, A_{x+y}, B_\alpha, B_{xy}, B_\beta]/\langle \mathcal{R}_2, \mathcal{R}_3, \mathcal{R}_4 \rangle \tag{488}$$

where the relations are

$$\mathcal{R}_2: \quad A_{c'}A_m, \quad A_{c'}A_{x+y}, \quad A_m^2, \tag{489a}$$

$$\mathcal{R}_3: \quad A_m A_{x+y}^2 + A_{x+y}^3 + A_{x+y}B_\alpha + A_m B_{xy}, \quad A_{x+y}(A_m A_{x+y} + A_{x+y}^2 + B_\alpha + B_{xy}), \quad A_m B_\beta, \quad A_{x+y}(A_m A_{x+y} + B_\beta), \tag{489b}$$

$$\mathcal{R}_4: \quad B_{xy}(B_\alpha + B_{xy}), \quad A_{c'}^2 B_\alpha + A_{c'}^2 B_\beta + B_\beta^2. \tag{489c}$$

We have the following table regarding IWPs and group cohomology at degree 3.

| Wyckoff position | Little group Intl. | Little group Schönflies | Coordinates | LSM anomaly class | Topo. inv. |
|---|---|---|---|---|---|
| 2a | 222 | $D_2$ | $(0,0,1/4)$, $(0,0,3/4)$ | $(A_{c'} + A_m)(A_{x+y}^2 + B_\alpha + B_{xy})$ | $\varphi_2[C_2, C_2']$ |
| 2b | 222 | $D_2$ | $(1/2,1/2,1/4)$, $(1/2,1/2,3/4)$ | $(A_{c'} + A_m)B_{xy}$ | $\varphi_2[T_1 T_2 C_2, T_1 T_2 C_2']$ |
| 2c | $\bar{4}$ | $S_4$ | $(0,0,0)$, $(0,0,1/2)$ | $A_m(A_{x+y}^2 + B_\alpha + B_{xy})$ | $\varphi_2[C_2'G, C_2]$ |
| 2d | $\bar{4}$ | $S_4$ | $(1/2,1/2,0)$, $(1/2,1/2,1/2)$ | $A_m B_{xy}$ | $\varphi_2[T_2 C_2'G, T_1 T_2 C_2]$ |
| 4i | 2 | $C_2$ | $(0,1/2,z)$, $(1/2,0,-z)$, $(0,1/2,z+1/2)$, $(1/2,0,-z+1/2)$ | $A_m A_{x+y}^2$ | $\varphi_2[G, T_1 C_2]$ |

**No. 117: $P\bar{4}b2$**

This group is generated by three translations $T_{1,2,3}$ as given in Eqs. (88), a two-fold rotation $C_2$, a two-fold rotation $C_2'$, and a glide $G$:

$$C_2: (x,y,z) \rightarrow (-x, -y, z), \tag{490a}$$

$$C_2': (x,y,z) \rightarrow (-y+1/2, -x+1/2, -z), \tag{490b}$$

$$G: (x,y,z) \rightarrow (x+1/2, -y+1/2, z). \tag{490c}$$

The $\mathbb{Z}_2$ cohomology ring is given by

$$\mathbb{Z}_2[A_{c'}, A_m, A_z, B_\alpha, B_\beta, C_\gamma]/\langle \mathcal{R}_2, \mathcal{R}_3, \mathcal{R}_4, \mathcal{R}_5, \mathcal{R}_6 \rangle \tag{491}$$

where the relations are

$$\mathcal{R}_2: \quad A_{c'}A_m, \quad A_m^2, \quad A_z(A_{c'} + A_z), \tag{492a}$$

$$\mathcal{R}_3: \quad A_m B_\alpha + A_{c'} B_\beta, \quad A_m(B_\alpha + B_\beta), \tag{492b}$$

$$\mathcal{R}_4: \quad A_m C_\gamma, \quad B_\beta(B_\alpha + B_\beta), \tag{492c}$$

$$\mathcal{R}_5: \quad B_\beta C_\gamma, \tag{492d}$$

$$\mathcal{R}_6: \quad A_{c'}^6 + A_{c'}^4 B_\alpha + B_\alpha^3 + B_\alpha^2 B_\beta + A_{c'} B_\alpha C_\gamma + C_\gamma^2. \tag{492e}$$

We have the following table regarding IWPs and group cohomology at degree 3.

| Wyckoff position | Little group Intl. | Little group Schönflies | Coordinates | LSM anomaly class | Topo. inv. |
|---|---|---|---|---|---|
| 2a | $\bar{4}$ | $S_4$ | $(0,0,0)$, $(1/2,1/2,0)$ | $A_m B_\alpha + A_z B_\beta$ | $\varphi_2[C_2'G, C_2]$ |
| 2b | $\bar{4}$ | $S_4$ | $(0,0,1/2)$, $(1/2,1/2,1/2)$ | $A_z B_\beta$ | $\varphi_2[T_3 C_2'G, C_2]$ |
| 2c | 222 | $D_2$ | $(0,1/2,0)$, $(1/2,0,0)$ | $A_{c'} B_\alpha + A_m B_\alpha + A_z B_\alpha + A_z B_\beta$ | $\varphi_2[T_1 C_2, C_2']$ |
| 2d | 222 | $D_2$ | $(0,1/2,1/2)$, $(1/2,0,1/2)$ | $A_z(B_\alpha + B_\beta)$ | $\varphi_2[T_1 C_2, T_3 C_2']$ |

**No. 118:** $P\bar{4}n2$

This group is generated by three translations $T_{1,2,3}$ as given in Eqs. (88), a two-fold rotation $C_2$, a two-fold rotation $C_2'$, and a glide $G$:

$$C_2\colon (x,y,z) \to (-x,-y,z), \tag{493a}$$
$$C_2'\colon (x,y,z) \to (-y+1/2, -x+1/2, -z+1/2), \tag{493b}$$
$$G\colon (x,y,z) \to (x+1/2, -y+1/2, z+1/2). \tag{493c}$$

The $\mathbb{Z}_2$ cohomology ring is given by

$$\mathbb{Z}_2[A_{c'}, A_m, A_{x+y+z}, B_\alpha, C_\gamma]/\langle \mathcal{R}_2, \mathcal{R}_3, \mathcal{R}_4, \mathcal{R}_5, \mathcal{R}_6\rangle \tag{494}$$

where the relations are

$$\mathcal{R}_2\colon \quad A_{c'}A_m, \quad A_m^2, \tag{495a}$$
$$\mathcal{R}_3\colon \quad A_m A_{x+y+z}^2, \quad A_{c'}^2 A_{x+y+z} + A_{c'}A_{x+y+z}^2 + A_{c'}B_\alpha + A_m B_\alpha, \tag{495b}$$
$$\mathcal{R}_4\colon \quad A_{x+y+z}(A_{c'} + A_{x+y+z})(A_{c'}A_{x+y+z} + A_{x+y+z}^2 + B_\alpha), \quad A_m C_\gamma, \tag{495c}$$
$$\mathcal{R}_5\colon \quad (A_{c'}A_{x+y+z} + A_{x+y+z}^2 + B_\alpha)C_\gamma, \tag{495d}$$
$$\mathcal{R}_6\colon \quad C_\gamma(A_{c'}^2 A_{x+y+z} + A_{c'}A_{x+y+z}^2 + C_\gamma). \tag{495e}$$

We have the following table regarding IWPs and group cohomology at degree 3.

| Wyckoff position | Little group Intl. | Little group Schönflies | Coordinates | LSM anomaly class | Topo. inv. |
|---|---|---|---|---|---|
| 2a | $\bar{4}$ | $S_4$ | $(0,0,0), (1/2,1/2,1/2)$ | $(A_{c'} + A_{x+y+z})(A_{c'}A_{x+y+z} + A_{x+y+z}^2 + B_\alpha)$ | $\varphi_2[C_2'G, C_2]$ |
| 2b | $\bar{4}$ | $S_4$ | $(0,0,1/2), (1/2,1/2,0)$ | $A_{x+y+z}(A_{c'}A_{x+y+z} + A_{x+y+z}^2 + B_\alpha)$ | $\varphi_2[T_3 C_2'G, C_2]$ |
| 2c | $222$ | $D_2$ | $(0,1/2,1/4), (1/2,0,3/4)$ | $A_{c'}^2 A_{x+y+z} + A_{c'}A_{x+y+z}^2 + C_\gamma$ | $\varphi_2[T_1 C_2, T_3 C_2']$ |
| 2d | $222$ | $D_2$ | $(0,1/2,3/4), (1/2,0,1/4)$ | $C_\gamma$ | $\varphi_2[T_1 C_2, C_2']$ |

**No. 119:** $I\bar{4}m2$

This group is generated by three translations $T_{1,2,3}$ as given in Eqs. (91), a two-fold rotation $C_2$, a two-fold rotation $C_2'$, and a mirror $M$:

$$C_2\colon (x,y,z) \to (-x,-y,z), \tag{496a}$$
$$C_2'\colon (x,y,z) \to (-y,-x,-z), \tag{496b}$$
$$M\colon (x,y,z) \to (x,-y,z). \tag{496c}$$

The $\mathbb{Z}_2$ cohomology ring is given by

$$\mathbb{Z}_2[A_{c'}, A_m, A_{x+y+z}, B_\alpha, B_\beta, B_{z(x+y)}, C_\gamma, C_{xyz}]/\langle \mathcal{R}_2, \mathcal{R}_3, \mathcal{R}_4, \mathcal{R}_5, \mathcal{R}_6\rangle \tag{497}$$

where the relations are

$$\mathcal{R}_2\colon \quad A_{c'}A_m, \quad A_m A_{x+y+z}, \quad A_{x+y+z}(A_{c'} + A_{x+y+z}), \tag{498a}$$
$$\mathcal{R}_3\colon \quad A_{x+y+z}B_\alpha + A_{c'}B_\beta, \quad A_{x+y+z}(B_\alpha + B_\beta), \quad A_{x+y+z}B_\alpha + A_{c'}B_{z(x+y)}, \quad A_{x+y+z}(B_\alpha + B_{z(x+y)}), \tag{498b}$$
$$\mathcal{R}_4\colon \quad A_{c'}C_\gamma + A_{x+y+z}C_\gamma + A_{c'}C_{xyz}, \quad A_{x+y+z}C_{xyz}, \quad B_\alpha B_\beta + B_\alpha B_{z(x+y)} + A_m C_\gamma, \quad B_\alpha B_\beta + B_\beta^2 + A_m C_\gamma,$$
$$\qquad B_\alpha B_\beta + B_\beta B_{z(x+y)} + A_m C_{xyz}, \quad B_\alpha B_\beta + B_{z(x+y)}^2 + A_m C_\gamma, \tag{498c}$$
$$\mathcal{R}_5\colon \quad (B_\beta + B_{z(x+y)})C_\gamma, \quad B_\alpha C_\gamma + B_\beta C_\gamma + B_\alpha C_{xyz}, \quad B_\beta C_{xyz}, \quad B_{z(x+y)}C_{xyz}, \tag{498d}$$
$$\mathcal{R}_6\colon \quad C_\gamma(A_{c'}B_\alpha + C_\gamma), \quad C_\gamma(A_{c'}B_\alpha + A_{x+y+z}B_\alpha + C_{xyz}), \quad A_{c'}B_\alpha C_\gamma + A_{x+y+z}B_\alpha C_\gamma + C_{xyz}^2. \tag{498e}$$

We have the following table regarding IWPs and group cohomology at degree 3.

| Wyckoff position | Little group Intl. | Little group Schönflies | Coordinates $(0,0,0) + (1/2,1/2,1/2)+$ | LSM anomaly class | Topo. inv. |
|---|---|---|---|---|---|
| 2a | $\bar{4}m2$ | $D_{2d}$ | $(0,0,0)$ | $A_{c'}B_\alpha + A_{x+y+z}B_\alpha + C_{xyz}$ | $\varphi_2[C_2, C_2']$ |
| 2b | $\bar{4}m2$ | $D_{2d}$ | $(0,0,1/2)$ | $C_{xyz}$ | $\varphi_2[C_2, T_1T_2C_2']$ |
| 2c | $\bar{4}m2$ | $D_{2d}$ | $(0,1/2,1/4)$ | $A_{x+y+z}B_\alpha + C_\gamma + C_{xyz}$ | $\varphi_2[T_1T_3C_2, T_1T_2T_3C_2']$ |
| 2d | $\bar{4}m2$ | $D_{2d}$ | $(0,1/2,3/4)$ | $C_\gamma + C_{xyz}$ | $\varphi_2[T_1T_3C_2, T_3C_2']$ |

### No. 120: $I\bar{4}c2$

This group is generated by three translations $T_{1,2,3}$ as given in Eqs. (91), a two-fold rotation $C_2$, a two-fold rotation $C_2'$, and a glide $G$:

$$C_2 : (x,y,z) \to (-x,-y,z), \tag{499a}$$
$$C_2' : (x,y,z) \to (-y,-x,-z+1/2), \tag{499b}$$
$$G : (x,y,z) \to (x,-y,z+1/2). \tag{499c}$$

The $\mathbb{Z}_2$ cohomology ring is given by

$$\mathbb{Z}_2[A_{c'}, A_m, A_{x+y+z}, B_\alpha, B_{z(x+y)}, D_\gamma, D_\delta]/\langle \mathcal{R}_2, \mathcal{R}_3, \mathcal{R}_4, \mathcal{R}_5, \mathcal{R}_6 \rangle \tag{500}$$

where the relations are

$$\mathcal{R}_2 : \quad A_{c'}A_m, \quad A_m A_{x+y+z}, \quad A_m^2 + A_{c'}A_{x+y+z} + A_{x+y+z}^2, \tag{501a}$$

$$\mathcal{R}_3 : \quad A_{x+y+z}B_\alpha + A_{c'}B_{z(x+y)}, \quad A_{x+y+z}(B_\alpha + B_{z(x+y)}), \tag{501b}$$

$$\mathcal{R}_4 : \quad B_{z(x+y)}(B_\alpha + B_{z(x+y)}), \tag{501c}$$

$$\mathcal{R}_5 : \quad A_m(B_\alpha^2 + B_\alpha B_{z(x+y)} + D_\gamma), \quad A_{x+y+z}D_\gamma + A_{c'}D_\delta, \quad A_m D_\delta, \quad A_{x+y+z}(D_\gamma + D_\delta), \tag{501d}$$

$$\mathcal{R}_6 : \quad B_{z(x+y)}D_\gamma + B_\alpha D_\delta, \quad B_{z(x+y)}(D_\gamma + D_\delta), \tag{501e}$$

$$\mathcal{R}_8 : \quad A_{c'}A_{x+y+z}B_\alpha^3 + B_\alpha^4 + B_\alpha^3 B_{z(x+y)} + A_{c'}^2 B_\alpha D_\gamma + D_\gamma^2, \quad A_{c'}A_{x+y+z}B_\alpha^3 + A_{c'}A_{x+y+z}B_\alpha D_\gamma + D_\gamma D_\delta,$$
$$A_{c'}A_{x+y+z}B_\alpha^3 + A_{c'}A_{x+y+z}B_\alpha D_\gamma + D_\delta^2. \tag{501f}$$

We have the following table regarding IWPs and group cohomology at degree 3.

| Wyckoff position | Little group Intl. | Little group Schönflies | Coordinates $(0,0,0) + (1/2,1/2,1/2)+$ | LSM anomaly class | Topo. inv. |
|---|---|---|---|---|---|
| 4a | 222 | $D_2$ | $(0,0,1/4), (0,0,3/4)$ | $(A_{c'} + A_m + A_{x+y+z})B_\alpha + A_m B_{z(x+y)}$ | $\varphi_2[C_2, C_2']$ |
| 4b | $\bar{4}$ | $S_4$ | $(0,0,0), (0,0,1/2)$ | $A_m(B_\alpha + B_{z(x+y)})$ | $\varphi_2[C_2'G, C_2]$ |
| 4c | $\bar{4}$ | $S_4$ | $(0,1/2,1/4), (0,1/2,3/4)$ | $A_m B_{z(x+y)}$ | $\varphi_2[T_1C_2'G, T_1T_3C_2]$ |
| 4d | 222 | $D_2$ | $(0,1/2,0), (1/2,0,0)$ | $A_{x+y+z}B_\alpha + A_m B_{z(x+y)}$ | $\varphi_2[T_1T_3C_2, T_3C_2']$ |

### No. 121: $I\bar{4}2m$

This group is generated by three translations $T_{1,2,3}$ as given in Eqs. (91), a two-fold rotation $C_2$, a two-fold rotation $C_2'$, and a mirror $M$:

$$C_2 : (x,y,z) \to (-x,-y,z), \tag{502a}$$
$$C_2' : (x,y,z) \to (-x,y,-z), \tag{502b}$$
$$M : (x,y,z) \to (y,x,z). \tag{502c}$$

The $\mathbb{Z}_2$ cohomology ring is given by

$$\mathbb{Z}_2[A_{c'}, A_m, A_{x+y+z}, B_\alpha, B_\beta, B_{z(x+y)}, C_{xyz}]/\langle \mathcal{R}_2, \mathcal{R}_3, \mathcal{R}_4, \mathcal{R}_5, \mathcal{R}_6 \rangle \tag{503}$$

where the relations are

$$\mathcal{R}_2: \quad A_{c'}A_m, \quad A_{c'}A_{x+y+z}, \quad A_{x+y+z}^2, \tag{504a}$$

$$\mathcal{R}_3: \quad A_{x+y+z}B_\alpha + A_m B_\beta, \quad A_{x+y+z}(B_\alpha + B_\beta), \quad A_{x+y+z}B_\alpha + A_m B_{z(x+y)}, \quad A_{x+y+z}(B_\alpha + B_{z(x+y)}), \tag{504b}$$

$$\mathcal{R}_4: \quad A_{x+y+z}C_{xyz}, \quad A_{c'}^2 B_\beta + B_\beta^2 + B_\alpha B_{z(x+y)}, \quad B_\alpha B_\beta + B_\beta B_{z(x+y)} + A_{c'}C_{xyz}, \quad B_{z(x+y)}(B_\alpha + B_{z(x+y)}), \tag{504c}$$

$$\mathcal{R}_5: \quad (A_{c'}^2 + B_\beta)C_{xyz}, \quad B_{z(x+y)}C_{xyz}, \tag{504d}$$

$$\mathcal{R}_6: \quad C_{xyz}(A_{c'}B_\alpha + C_{xyz}). \tag{504e}$$

We have the following table regarding IWPs and group cohomology at degree 3.

| Wyckoff position | Little group Intl. | Little group Schönflies | Coordinates $(0,0,0) + (1/2,1/2,1/2)+$ | LSM anomaly class | Topo. inv. |
|---|---|---|---|---|---|
| 2a | $\overline{4}2m$ | $D_{2d}$ | $(0,0,0)$ | $A_{c'}B_\alpha + A_{c'}B_{z(x+y)} + C_{xyz}$ | $\varphi_2[C_2, C_2']$ |
| 2b | $\overline{4}2m$ | $D_{2d}$ | $(0,0,1/2)$ | $C_{xyz}$ | $\varphi_2[C_2, T_1 T_2 C_2']$ |
| 4c | $222$ | $D_2$ | $(0,1/2,0), (1/2,0,0)$ | $A_{x+y+z}B_\alpha + A_{c'}B_{z(x+y)}$ | $\varphi_2[T_1 T_3 C_2, C_2']$ |
| 4d | $\overline{4}$ | $S_4$ | $(0,1/2,1/4), (0,1/2,3/4)$ | $A_{x+y+z}B_\alpha$ | $\varphi_2[T_1 T_2 T_3 C_2' M, T_1 T_3 C_2]$ |

### No. 122: $I\overline{4}2d$

This group is generated by three translations $T_{1,2,3}$ as given in Eqs. (91), a two-fold rotation $C_2$, a two-fold rotation $C_2'$, and a glide $G$:

$$C_2: (x,y,z) \to (-x,-y,z), \tag{505a}$$

$$C_2': (x,y,z) \to (-x+1/2, y, -z+3/4), \tag{505b}$$

$$G: (x,y,z) \to (y, x+1/2, z+1/4). \tag{505c}$$

The $\mathbb{Z}_2$ cohomology ring is given by

$$\mathbb{Z}_2[A_{c'}, A_m, B_\alpha, B_{z(x+y)}, C_\gamma]/\langle \mathcal{R}_2, \mathcal{R}_3, \mathcal{R}_4, \mathcal{R}_5, \mathcal{R}_6\rangle \tag{506}$$

where the relations are

$$\mathcal{R}_2: \quad A_{c'}A_m, \quad A_m^2, \tag{507a}$$

$$\mathcal{R}_3: \quad (A_{c'} + A_m)B_\alpha, \quad A_{c'}B_\alpha + A_m B_{z(x+y)}, \tag{507b}$$

$$\mathcal{R}_4: \quad (A_{c'} + A_m)C_\gamma, \quad B_\alpha^2 + B_\alpha B_{z(x+y)} + A_{c'}C_\gamma, \quad (B_\alpha + B_{z(x+y)})^2, \tag{507c}$$

$$\mathcal{R}_5: \quad (B_\alpha + B_{z(x+y)})C_\gamma, \tag{507d}$$

$$\mathcal{R}_6: \quad C_\gamma(A_{c'}B_\alpha + C_\gamma). \tag{507e}$$

We have the following table regarding IWPs and group cohomology at degree 3.

| Wyckoff position | Little group Intl. | Little group Schönflies | Coordinates $(0,0,0) + (1/2,1/2,1/2)+$ | LSM anomaly class | Topo. inv. |
|---|---|---|---|---|---|
| 4a | $\overline{4}$ | $S_4$ | $(0,0,0), (1/2,0,3/4)$ | $C_\gamma$ | $\varphi_2[C_2'G, T_1 T_3 C_2]$ |
| 4b | $\overline{4}$ | $S_4$ | $(0,0,1/2), (1/2,0,1/4)$ | $A_{c'}B_\alpha + C_\gamma$ | $\varphi_2[T_3^{-1}C_2'G, C_2]$ |
| 8c | $2$ | $C_2$ | $(0,0,z), (0,0,-z),$ $(1/2,0,-z+3/4), (1/2,0,z+3/4)$ | $A_{c'}(B_\alpha + B_{z(x+y)})$ | $\varphi_2[T_2 T_3, T_3 C_2 C_2']$ |

### No. 123: $P4/mmm$

This group is generated by three translations $T_{1,2,3}$ as given in Eqs. (88), a two-fold rotation $C_2$, a two-fold rotation $C_2'$, a mirror $M$, and an inversion $I$:

$$C_2: (x,y,z) \to (-x,-y,z), \tag{508a}$$

$$C_2': (x,y,z) \to (-x, y, -z), \tag{508b}$$

$$M: (x,y,z) \to (y,x,z), \tag{508c}$$

$$I: (x,y,z) \to (-x,-y,-z). \tag{508d}$$

The $\mathbb{Z}_2$ cohomology ring is given by

$$\mathbb{Z}_2[A_i, A_m, A_{c'}, A_{x+y}, A_z, B_\alpha, B_{xy}]/\langle \mathscr{R}_2, \mathscr{R}_3, \mathscr{R}_4 \rangle \tag{509}$$

where the relations are

$$\mathscr{R}_2: \quad A_{c'}A_m, \quad A_m A_{x+y}, \quad A_z(A_{c'} + A_i + A_z), \tag{510a}$$

$$\mathscr{R}_3: \quad A_{c'}A_{x+y}^2 + A_{x+y}^3 + A_{x+y}B_\alpha + A_{c'}B_{xy}, \quad A_{x+y}(A_{c'}A_{x+y} + A_{x+y}^2 + B_\alpha + B_{xy}), \tag{510b}$$

$$\mathscr{R}_4: \quad B_{xy}(B_\alpha + B_{xy}). \tag{510c}$$

We have the following table regarding IWPs and group cohomology at degree 3.

| Wyckoff position | Little group | | Coordinates | LSM anomaly class | Topo. inv. |
|---|---|---|---|---|---|
| | Intl. | Schönflies | | | |
| 1a | $4/mmm$ | $D_{4h}$ | $(0,0,0)$ | $(A_{c'} + A_i + A_z)(A_{c'}A_{x+y} + A_{x+y}^2 + B_\alpha + B_{xy})$ | $\varphi_2[C_2, C_2']$ |
| 1b | $4/mmm$ | $D_{4h}$ | $(0,0,1/2)$ | $A_z(A_{c'}A_{x+y} + A_{x+y}^2 + B_\alpha + B_{xy})$ | $\varphi_2[C_2, T_3C_2']$ |
| 1c | $4/mmm$ | $D_{4h}$ | $(1/2,1/2,0)$ | $(A_{c'} + A_i + A_z)B_{xy}$ | $\varphi_2[T_1T_2C_2, T_1C_2']$ |
| 1d | $4/mmm$ | $D_{4h}$ | $(1/2,1/2,1/2)$ | $A_z B_{xy}$ | $\varphi_2[T_1T_2C_2, T_1T_3C_2']$ |
| 2e | $mmm$ | $D_{2h}$ | $(0,1/2,1/2), (1/2,0,1/2)$ | $A_{x+y}(A_{c'} + A_{x+y})A_z$ | $\varphi_2[T_1C_2, T_1T_3C_2']$ |
| 2f | $mmm$ | $D_{2h}$ | $(0,1/2,0), (1/2,0,0)$ | $A_{x+y}(A_{c'} + A_{x+y})(A_{c'} + A_i + A_z)$ | $\varphi_2[T_1C_2, T_1C_2']$ |

## No. 124: $P4/mcc$

This group is generated by three translations $T_{1,2,3}$ as given in Eqs. (88), a two-fold rotation $C_2$, a two-fold rotation $C_2'$, a glide $G$, and an inversion $I$:

$$C_2: (x,y,z) \to (-x,-y,z), \tag{511a}$$

$$C_2': (x,y,z) \to (-x,y,-z+1/2), \tag{511b}$$

$$G: (x,y,z) \to (y,x,z+1/2), \tag{511c}$$

$$I: (x,y,z) \to (-x,-y,-z). \tag{511d}$$

The $\mathbb{Z}_2$ cohomology ring is given by

$$\mathbb{Z}_2[A_i, A_m, A_{c'}, A_{x+y}, B_\alpha, B_{xy}]/\langle \mathscr{R}_2, \mathscr{R}_3, \mathscr{R}_4 \rangle \tag{512}$$

where the relations are

$$\mathscr{R}_2: \quad A_{c'}A_m, \quad A_m A_{x+y}, \quad A_{c'}A_i + A_i A_m + A_m^2, \tag{513a}$$

$$\mathscr{R}_3: \quad A_{c'}A_{x+y}^2 + A_{x+y}^3 + A_{x+y}B_\alpha + A_{c'}B_{xy}, \quad A_{x+y}(A_{c'}A_{x+y} + A_{x+y}^2 + B_\alpha + B_{xy}), \tag{513b}$$

$$\mathscr{R}_4: \quad B_{xy}(B_\alpha + B_{xy}). \tag{513c}$$

We have the following table regarding IWPs and group cohomology at degree 3.

| Wyckoff position | Little group | | Coordinates | LSM anomaly class | Topo. inv. |
|---|---|---|---|---|---|
| | Intl. | Schönflies | | | |
| 2a | $422$ | $D_4$ | $(0,0,1/4), (0,0,3/4)$ | $(A_{c'} + A_m)(A_{c'}A_{x+y} + A_{x+y}^2 + B_\alpha + B_{xy})$ | $\varphi_2[C_2, GI]$ |
| 2b | $4/m$ | $C_{4h}$ | $(0,0,0), (0,0,1/2)$ | $A_i A_{x+y}^2 + (A_i + A_m)(B_\alpha + B_{xy})$ | $\varphi_1[I]$ |
| 2c | $422$ | $D_4$ | $(1/2,1/2,1/4), (1/2,1/2,3/4)$ | $(A_{c'} + A_m)B_{xy}$ | $\varphi_2[T_1T_2C_2, T_1T_2GI]$ |
| 2d | $4/m$ | $C_{4h}$ | $(1/2,1/2,0), (1/2,1/2,1/2)$ | $(A_i + A_m)B_{xy}$ | $\varphi_1[T_1T_2I]$ |
| 4e | $2/m$ | $C_{2h}$ | $(0,1/2,0), (1/2,0,0),$ $(0,1/2,1/2), (1/2,0,1/2)$ | $A_i A_{x+y}^2$ | $\varphi_1[T_1I]$ |
| 4f | $222$ | $D_2$ | $(0,1/2,1/4), (1/2,0,1/4),$ $(0,1/2,3/4), (1/2,0,3/4)$ | $A_{c'}A_{x+y}(A_{c'} + A_{x+y})$ | $\varphi_2[T_1C_2, T_1C_2']$ |

**No. 125:** $P4/nbm$

This group is generated by three translations $T_{1,2,3}$ as given in Eqs. (88), a two-fold rotation $C_2$, a two-fold rotation $C_2'$, a glide $G$, and an inversion $I$:

$$C_2\colon (x,y,z) \to (-x+1/2, -y+1/2, z), \tag{514a}$$
$$C_2'\colon (x,y,z) \to (-x+1/2, y, -z), \tag{514b}$$
$$G\colon (x,y,z) \to (y+1/2, x+1/2, z), \tag{514c}$$
$$I\colon (x,y,z) \to (-x, -y, -z). \tag{514d}$$

The $\mathbb{Z}_2$ cohomology ring is given by

$$\mathbb{Z}_2[A_i, A_m, A_{c'}, A_z, B_\alpha, B_\beta]/\langle \mathcal{R}_2, \mathcal{R}_3, \mathcal{R}_4 \rangle \tag{515}$$

where the relations are

$$\mathcal{R}_2\colon \quad A_{c'}A_i, \quad A_{c'}A_m, \quad A_z(A_{c'} + A_i + A_z), \tag{516a}$$
$$\mathcal{R}_3\colon \quad A_i B_\beta, \quad A_i^3 + A_i A_m^2 + A_i B_\alpha + A_m B_\alpha + A_m B_\beta, \tag{516b}$$
$$\mathcal{R}_4\colon \quad B_\beta(B_\alpha + B_\beta). \tag{516c}$$

We have the following table regarding IWPs and group cohomology at degree 3.

| Wyckoff position | Little group Intl. | Little group Schönflies | Coordinates | LSM anomaly class | Topo. inv. |
|---|---|---|---|---|---|
| 2a | 422 | $D_4$ | $(1/4, 1/4, 0)$, $(3/4, 3/4, 0)$ | $(A_{c'} + A_i + A_z)(A_i^2 + A_i A_m + B_\alpha + B_\beta)$ | $\varphi_2[C_2, C_2']$ |
| 2b | 422 | $D_4$ | $(1/4, 1/4, 1/2)$, $(3/4, 3/4, 1/2)$ | $A_z(A_i^2 + A_i A_m + B_\alpha + B_\beta)$ | $\varphi_2[C_2, T_3 C_2']$ |
| 2c | $\overline{4}2m$ | $D_{2d}$ | $(3/4, 1/4, 0)$, $(1/4, 3/4, 0)$ | $(A_{c'} + A_z)B_\beta$ | $\varphi_2[T_2 C_2, C_2']$ |
| 2d | $\overline{4}2m$ | $D_{2d}$ | $(3/4, 1/4, 1/2)$, $(1/4, 3/4, 1/2)$ | $A_z B_\beta$ | $\varphi_2[T_2 C_2, T_3 C_2']$ |
| 4e | $2/m$ | $C_{2h}$ | $(0,0,0)$, $(1/2,1/2,0)$, $(1/2,0,0)$, $(0,1/2,0)$ | $A_i(A_i + A_m)(A_i + A_z)$ | $\varphi_1[I]$ |
| 4f | $2/m$ | $C_{2h}$ | $(0,0,1/2)$, $(1/2,1/2,1/2)$, $(1/2,0,1/2)$, $(0,1/2,1/2)$ | $A_i(A_i + A_m)A_z$ | $\varphi_1[T_3 I]$ |

**No. 126:** $P4/nnc$

This group is generated by three translations $T_{1,2,3}$ as given in Eqs. (88), a two-fold rotation $C_2$, a two-fold rotation $C_2'$, a glide $G$, and an inversion $I$:

$$C_2\colon (x,y,z) \to (-x+1/2, -y+1/2, z), \tag{517a}$$
$$C_2'\colon (x,y,z) \to (-x+1/2, y, -z+1/2), \tag{517b}$$
$$G\colon (x,y,z) \to (y+1/2, x+1/2, z+1/2), \tag{517c}$$
$$I\colon (x,y,z) \to (-x, -y, -z). \tag{517d}$$

The $\mathbb{Z}_2$ cohomology ring is given by

$$\mathbb{Z}_2[A_i, A_{c'}, A_m, B_\alpha, B_{\beta 1}, B_{\beta 2}, C_\gamma]/\langle \mathcal{R}_2, \mathcal{R}_3, \mathcal{R}_4, \mathcal{R}_5, \mathcal{R}_6 \rangle \tag{518}$$

where the relations are

$$\mathcal{R}_2\colon \quad A_{c'}A_i, \quad A_{c'}A_m, \quad A_m(A_i + A_m), \tag{519a}$$
$$\mathcal{R}_3\colon \quad A_i B_{\beta 1}, \quad A_i B_\alpha + A_m B_\alpha + A_m B_{\beta 1}, \quad A_i B_{\beta 2}, \quad A_m B_{\beta 2}, \tag{519b}$$
$$\mathcal{R}_4\colon \quad (A_i + A_m)C_\gamma, \quad B_{\beta 1}(A_{c'}^2 + B_\alpha + B_{\beta 1}), \quad A_{c'}^2 B_{\beta 2} + B_\alpha B_{\beta 2} + B_{\beta 1}B_{\beta 2} + A_{c'}C_\gamma, \quad A_{c'}^2 B_{\beta 1} + A_{c'}^2 B_{\beta 2} + B_{\beta 2}^2, \tag{519c}$$
$$\mathcal{R}_5\colon \quad B_{\beta 1}C_\gamma, \quad (A_{c'}^2 + B_{\beta 2})C_\gamma, \tag{519d}$$
$$\mathcal{R}_6\colon \quad C_\gamma(A_{c'}^3 + A_{c'}B_\alpha + A_i B_\alpha + C_\gamma). \tag{519e}$$

We have the following table regarding IWPs and group cohomology at degree 3.

| Wyckoff position | Little group Intl. | Little group Schönflies | Coordinates | LSM anomaly class | Topo. inv. |
|---|---|---|---|---|---|
| 2a | 422 | $D_4$ | $(1/4,1/4,1/4)$, $(3/4,3/4,3/4)$ | $A_{c'}^3 + A_{c'}B_\alpha + A_iB_\alpha + A_{c'}B_{\beta 1} + C_\gamma$ | $\varphi_2[C_2, C_2']$ |
| 2b | 422 | $D_4$ | $(1/4,1/4,3/4)$, $(3/4,3/4,1/4)$ | $C_\gamma$ | $\varphi_2[C_2, T_3C_2']$ |
| 4c | 222 | $D_2$ | $(1/4,3/4,3/4)$, $(3/4,1/4,3/4)$, $(3/4,1/4,1/4)$, $(1/4,3/4,1/4)$ | $A_iB_\alpha + A_mB_\alpha + A_{c'}B_{\beta 1}$ | $\varphi_2[T_1C_2, T_1T_3C_2']$ |
| 4d | $\bar{4}$ | $S_4$ | $(1/4,3/4,0)$, $(3/4,1/4,0)$, $(1/4,3/4,1/2)$, $(3/4,1/4,1/2)$ | $(A_i + A_m)B_\alpha$ | $\varphi_2[C_2'G, T_1^{-1}C_2]$ |
| 8f | $\bar{1}$ | $C_i$ | $(0,0,0)$, $(1/2,1/2,0)$, $(1/2,0,0)$, $(0,1/2,0)$, $(1/2,0,1/2)$, $(0,1/2,1/2)$, $(0,0,1/2)$, $(1/2,1/2,1/2)$ | $A_i^2(A_i + A_m)$ | $\varphi_1[I]$ |

**No. 127:** $P4/mbm$

This group is generated by three translations $T_{1,2,3}$ as given in Eqs. (88), a two-fold rotation $C_2$, a two-fold screw $S_2'$, a glide $G$, and an inversion $I$:

$$C_2 : (x,y,z) \to (-x,-y,z), \tag{520a}$$
$$S_2' : (x,y,z) \to (-x+1/2, y+1/2, -z), \tag{520b}$$
$$G : (x,y,z) \to (y+1/2, x+1/2, z), \tag{520c}$$
$$I : (x,y,z) \to (-x,-y,-z). \tag{520d}$$

The $\mathbb{Z}_2$ cohomology ring is given by

$$\mathbb{Z}_2[A_i, A_m, A_{c'}, A_z, B_\alpha, B_\beta, C_\gamma]/\langle \mathscr{R}_2, \mathscr{R}_3, \mathscr{R}_4, \mathscr{R}_5, \mathscr{R}_6 \rangle \tag{521}$$

where the relations are

$$\mathscr{R}_2 : \quad A_{c'}A_m, \quad A_{c'}^2, \quad A_z(A_{c'} + A_i + A_z), \tag{522a}$$
$$\mathscr{R}_3 : \quad A_{c'}B_\alpha + A_mB_\beta, \quad A_{c'}(B_\alpha + B_\beta), \tag{522b}$$
$$\mathscr{R}_4 : \quad A_{c'}(A_iB_\alpha + C_\gamma), \quad B_\beta(B_\alpha + B_\beta), \tag{522c}$$
$$\mathscr{R}_5 : \quad A_{c'}B_\alpha^2 + A_iB_\alpha B_\beta + B_\beta C_\gamma, \tag{522d}$$
$$\mathscr{R}_6 : \quad A_i^4A_m^2 + A_m^6 + A_i^2A_m^2B_\alpha + A_m^4B_\alpha + A_{c'}A_iB_\alpha^2 + B_\alpha^3 + A_i^2B_\alpha B_\beta + B_\alpha^2B_\beta + A_mB_\alpha C_\gamma + C_\gamma^2. \tag{522e}$$

We have the following table regarding IWPs and group cohomology at degree 3.

| Wyckoff position | Little group Intl. | Little group Schönflies | Coordinates | LSM anomaly class | Topo. inv. |
|---|---|---|---|---|---|
| 2a | $4/m$ | $C_{4h}$ | $(0,0,0)$, $(1/2,1/2,0)$ | $A_{c'}B_\alpha + A_iB_\beta + A_zB_\beta$ | $\varphi_1[I]$ |
| 2b | $4/m$ | $C_{4h}$ | $(0,0,1/2)$, $(1/2,1/2,1/2)$ | $A_zB_\beta$ | $\varphi_1[T_3I]$ |
| 2c | $mmm$ | $D_{2h}$ | $(0,1/2,1/2)$, $(1/2,0,1/2)$ | $A_z(B_\alpha + B_\beta)$ | $\varphi_2[T_2C_2, T_3GI]$ |
| 2d | $mmm$ | $D_{2h}$ | $(0,1/2,0)$, $(1/2,0,0)$ | $(A_i + A_z)(B_\alpha + B_\beta)$ | $\varphi_2[T_2C_2, GI]$ |

**No. 128:** $P4/mnc$

This group is generated by three translations $T_{1,2,3}$ as given in Eqs. (88), a two-fold rotation $C_2$, a two-fold screw $S_2'$, a glide $G$, and an inversion $I$:

$$C_2 : (x,y,z) \to (-x,-y,z), \tag{523a}$$
$$S_2' : (x,y,z) \to (-x+1/2, y+1/2, -z+1/2), \tag{523b}$$
$$G : (x,y,z) \to (y+1/2, x+1/2, z+1/2), \tag{523c}$$
$$I : (x,y,z) \to (-x,-y,-z). \tag{523d}$$

The $\mathbb{Z}_2$ cohomology ring is given by

$$\mathbb{Z}_2[A_i, A_m, A_{c'}, B_\alpha, B_{\beta1}, B_{\beta2}, C_{\gamma1}, C_{\gamma2}]/\langle \mathcal{R}_2, \mathcal{R}_3, \mathcal{R}_4, \mathcal{R}_5, \mathcal{R}_6 \rangle \tag{524}$$

where the relations are

$$\mathcal{R}_2: \quad A_{c'}A_m, \quad A_{c'}A_i + A_iA_m + A_m^2, \quad A_{c'}^2, \tag{525a}$$

$$\mathcal{R}_3: \quad A_{c'}B_\alpha + A_mB_{\beta1}, \quad A_{c'}(B_\alpha + B_{\beta1}), \quad A_{c'}B_\alpha + A_mB_\alpha + A_mB_{\beta2}, \quad A_{c'}B_{\beta2}, \tag{525b}$$

$$\mathcal{R}_4: \quad (A_{c'} + A_m)C_{\gamma1}, \quad A_mC_{\gamma1} + A_{c'}C_{\gamma2},$$
$$A_i^3A_m + A_i^2A_m^2 + A_iA_mB_\alpha + B_\alpha^2 + B_\alpha B_{\beta1} + A_i^2B_{\beta2} + B_\alpha B_{\beta2} + A_iC_{\gamma1} + A_iC_{\gamma2} + A_mC_{\gamma2}, \quad B_{\beta1}(B_\alpha + B_{\beta1}),$$
$$B_{\beta1}B_{\beta2} + A_mC_{\gamma1}, \quad A_{c'}A_iB_\alpha + A_i^2B_\alpha + A_iA_mB_\alpha + B_\alpha^2 + A_i^2B_{\beta1} + B_\alpha B_{\beta1} + B_{\beta2}^2, \tag{525c}$$

$$\mathcal{R}_5: \quad (B_\alpha + B_{\beta1})C_{\gamma1}, \quad (A_iA_m + B_{\beta2})C_{\gamma1}, \quad A_{c'}A_i^2B_\alpha + A_{c'}B_\alpha^2 + A_iA_mC_{\gamma1} + B_\alpha C_{\gamma1} + B_{\beta1}C_{\gamma2},$$
$$A_i^4A_m + A_i^3A_m^2 + A_i^3B_\alpha + A_iB_\alpha^2 + A_mB_\alpha^2 + A_i^3B_{\beta1} + A_iB_\alpha B_{\beta1} + A_i^3B_{\beta2} + A_i^2C_{\gamma1} + A_iA_mC_{\gamma1}$$
$$+ B_\alpha C_{\gamma1} + A_i^2C_{\gamma2} + A_iA_mC_{\gamma2} + B_\alpha C_{\gamma2} + B_{\beta2}C_{\gamma2}, \tag{525d}$$

$$\mathcal{R}_6: \quad C_{\gamma1}(A_iB_\alpha + A_mB_\alpha + C_{\gamma1}), \quad C_{\gamma1}(A_iB_\alpha + C_{\gamma2}),$$
$$A_i^4A_m^2 + A_i^4B_\alpha + A_{c'}A_iB_\alpha^2 + A_i^2B_\alpha^2 + A_iA_mB_\alpha^2 + B_\alpha^3 + A_i^4B_{\beta1} + A_i^2B_\alpha B_{\beta1} + B_\alpha^2B_{\beta1} + A_iB_\alpha C_{\gamma1}$$
$$+ A_mB_\alpha C_{\gamma2} + C_{\gamma2}^2. \tag{525e}$$

We have the following table regarding IWPs and group cohomology at degree 3.

| Wyckoff position | Little group Intl. | Little group Schönflies | Coordinates | LSM anomaly class | Topo. inv. |
|---|---|---|---|---|---|
| 2a | $4/m$ | $C_{4h}$ | $(0,0,0)$, $(1/2,1/2,1/2)$ | $A_{c'}B_\alpha + A_iB_{\beta1} + C_{\gamma1}$ | $\varphi_1[I]$ |
| 2b | $4/m$ | $C_{4h}$ | $(0,0,1/2)$, $(1/2,1/2,0)$ | $C_{\gamma1}$ | $\varphi_1[T_3I]$ |
| 4c | $2/m$ | $C_{2h}$ | $(0,1/2,0)$, $(1/2,0,0)$, $(1/2,0,1/2)$, $(0,1/2,1/2)$ | $A_{c'}B_\alpha + A_iB_\alpha + A_mB_\alpha + A_iB_{\beta1}$ | $\varphi_1[T_1I]$ |
| 4d | $222$ | $D_2$ | $(0,1/2,1/4)$, $(1/2,0,1/4)$, $(0,1/2,3/4)$, $(1/2,0,3/4)$ | $(A_{c'} + A_m)B_\alpha$ | $\varphi_2[T_1C_2, GI]$ |

**No. 129:** $P4/nmm$

This group is generated by three translations $T_{1,2,3}$ as given in Eqs. (88), a two-fold rotation $C_2$, a two-fold screw $S_2'$, a mirror $M$, and an inversion $I$:

$$C_2: (x,y,z) \rightarrow (-x+1/2, -y+1/2, z), \tag{526a}$$

$$S_2': (x,y,z) \rightarrow (-x, y+1/2, -z), \tag{526b}$$

$$M: (x,y,z) \rightarrow (y,x,z), \tag{526c}$$

$$I: (x,y,z) \rightarrow (-x,-y,-z). \tag{526d}$$

The $\mathbb{Z}_2$ cohomology ring is given by

$$\mathbb{Z}_2[A_i, A_m, A_{c'}, A_z, B_\alpha, B_\beta]/\langle \mathcal{R}_2, \mathcal{R}_3, \mathcal{R}_4 \rangle \tag{527}$$

where the relations are

$$\mathcal{R}_2: \quad A_{c'}A_m, \quad A_{c'}(A_{c'} + A_i), \quad A_z(A_{c'} + A_i + A_z), \tag{528a}$$

$$\mathcal{R}_3: \quad A_{c'}A_i^2 + A_i^3 + A_i^2A_m + A_{c'}B_\alpha + A_iB_\alpha + A_mB_\beta, \quad A_{c'}A_i^2 + A_i^3 + A_i^2A_m + A_{c'}B_\alpha + A_iB_\alpha + A_{c'}B_\beta + A_iB_\beta, \tag{528b}$$

$$\mathcal{R}_4: \quad B_\beta(B_\alpha + B_\beta). \tag{528c}$$

We have the following table regarding IWPs and group cohomology at degree 3.

| Wyckoff position | Little group Intl. | Little group Schönflies | Coordinates | LSM anomaly class | Topo. inv. |
|---|---|---|---|---|---|
| 2a | $\bar{4}m2$ | $D_{2d}$ | $(3/4,1/4,0)$, $(1/4,3/4,0)$ | $(A_{c'} + A_i + A_z)B_\beta$ | $\varphi_2[T_2C_2, T_1T_2MI]$ |
| 2b | $\bar{4}m2$ | $D_{2d}$ | $(3/4,1/4,1/2)$, $(1/4,3/4,1/2)$ | $A_z B_\beta$ | $\varphi_2[T_2C_2, T_1T_2T_3MI]$ |
| 2c | $4mm$ | $C_{4v}$ | $(1/4,1/4,z)$, $(3/4,3/4,-z)$ | $A_z(A_{c'}A_i + A_i^2 + A_iA_m + B_\alpha + B_\beta)$ | $\varphi_2[T_3, C_2]$ |
| 4d | $2/m$ | $C_{2h}$ | $(0,0,0)$, $(1/2,1/2,0)$, $(1/2,0,0)$, $(0,1/2,0)$ | $A_i(A_{c'} + A_i + A_m)(A_i + A_z)$ | $\varphi_1[I]$ |
| 4e | $2/m$ | $C_{2h}$ | $(0,0,1/2)$, $(1/2,1/2,1/2)$, $(1/2,0,1/2)$, $(0,1/2,1/2)$ | $A_i(A_{c'} + A_i + A_m)A_z$ | $\varphi_1[T_3I]$ |

**No. 130:** $P4/ncc$

This group is generated by three translations $T_{1,2,3}$ as given in Eqs. (88), a two-fold rotation $C_2$, a two-fold screw $S_2'$, a glide $G$, and an inversion $I$:

$$C_2 \colon (x,y,z) \to (-x+1/2, -y+1/2, z), \tag{529a}$$
$$S_2' \colon (x,y,z) \to (-x, y+1/2, -z+1/2), \tag{529b}$$
$$G \colon (x,y,z) \to (y, x, z+1/2), \tag{529c}$$
$$I \colon (x,y,z) \to (-x, -y, -z). \tag{529d}$$

The $\mathbb{Z}_2$ cohomology ring is given by

$$\mathbb{Z}_2[A_i, A_m, A_{c'}, B_\alpha, B_\beta, D_\gamma]/\langle \mathcal{R}_2, \mathcal{R}_3, \mathcal{R}_4, \mathcal{R}_5, \mathcal{R}_6 \rangle \tag{530}$$

where the relations are

$$\mathcal{R}_2 \colon \quad A_{c'}A_m, \quad A_{c'}A_i + A_iA_m + A_m^2, \quad A_{c'}(A_{c'} + A_i), \tag{531a}$$
$$\mathcal{R}_3 \colon \quad A_i^3 + A_i^2A_m + A_{c'}B_\alpha + A_iB_\alpha + A_mB_\beta, \quad A_i^3 + A_i^2A_m + A_{c'}B_\alpha + A_iB_\alpha + A_{c'}B_\beta + A_iB_\beta, \tag{531b}$$
$$\mathcal{R}_4 \colon \quad B_\beta(B_\alpha + B_\beta), \tag{531c}$$
$$\mathcal{R}_5 \colon \quad (A_i + A_m)D_\gamma, \quad A_{c'}D_\gamma, \tag{531d}$$
$$\mathcal{R}_6 \colon \quad (B_\alpha + B_\beta)D_\gamma, \tag{531e}$$
$$\mathcal{R}_8 \colon \quad A_i^8 + A_i^6B_\alpha + A_iA_mB_\alpha^3 + A_i^2B_\alpha D_\gamma + D_\gamma^2. \tag{531f}$$

We have the following table regarding IWPs and group cohomology at degree 3.

| Wyckoff position | Little group Intl. | Little group Schönflies | Coordinates | LSM anomaly class | Topo. inv. |
|---|---|---|---|---|---|
| 4a | $222$ | $D_2$ | $(3/4,1/4,1/4)$, $(1/4,3/4,1/4)$, $(1/4,3/4,3/4)$, $(3/4,1/4,3/4)$ | $A_iB_\beta$ | $\varphi_2[T_2C_2, T_1T_2GI]$ |
| 4b | $\bar{4}$ | $S_4$ | $(3/4,1/4,0)$, $(1/4,3/4,0)$, $(1/4,3/4,1/2)$, $(3/4,1/4,1/2)$ | $A_i^3 + A_i^2A_m + A_{c'}B_\alpha + A_iB_\alpha + A_iB_\beta$ | $\varphi_2[S_2'G, T_1^{-1}C_2]$ |
| 4c | $4$ | $C_4$ | $(1/4,1/4,z)$, $(3/4,3/4,-z+1/2)$, $(3/4,3/4,-z)$, $(1/4,1/4,z+1/2)$ | $A_{c'}B_\alpha + A_mB_\alpha + A_iB_\beta$ | $\varphi_2[G, C_2]$ |
| 8d | $\bar{1}$ | $C_i$ | $(0,0,0)$, $(1/2,1/2,0)$, $(1/2,0,0)$, $(0,1/2,0)$, $(0,1/2,1/2)$, $(1/2,0,1/2)$, $(1/2,1/2,1/2)$, $(0,0,1/2)$ | $A_i^2(A_i + A_m)$ | $\varphi_1[I]$ |

**No. 131:** $P4_2/mmc$

This group is generated by three translations $T_{1,2,3}$ as given in Eqs. (88), a two-fold rotation $C_2$, a two-fold rotation $C_2'$, a glide $G$, and an inversion $I$:

$$C_2 \colon (x,y,z) \to (-x,-y,z), \tag{532a}$$
$$C_2' \colon (x,y,z) \to (-x,y,-z), \tag{532b}$$
$$G \colon (x,y,z) \to (y,x,z+1/2), \tag{532c}$$
$$I \colon (x,y,z) \to (-x,-y,-z). \tag{532d}$$

The $\mathbb{Z}_2$ cohomology ring is given by

$$\mathbb{Z}_2[A_i, A_m, A_{c'}, A_{x+y}, B_\alpha, B_\beta, B_{xy}, B_{z(x+y)}]/\langle \mathscr{R}_2, \mathscr{R}_3, \mathscr{R}_4 \rangle \tag{533}$$

where the relations are

$$\mathscr{R}_2 \colon \quad A_{c'}A_m, \quad A_m A_{x+y}, \quad A_m(A_i + A_m), \tag{534a}$$

$$\mathscr{R}_3 \colon \quad A_m B_\beta, \quad A_{c'}A_{x+y}^2 + A_{x+y}^3 + A_{x+y}B_\alpha + A_{c'}B_{xy}, \quad A_{x+y}(A_{c'}A_{x+y} + A_{x+y}^2 + B_\alpha + B_{xy}), \quad A_m B_{z(x+y)},$$
$$A_{c'}A_i A_{x+y} + A_i^2 A_{x+y} + A_i A_{x+y}^2 + A_{x+y}B_\beta + A_{c'}B_{z(x+y)}, \tag{534b}$$

$$\mathscr{R}_4 \colon \quad B_{xy}(B_\alpha + B_{xy}), \quad A_{c'}A_i A_{x+y}^2 + A_i^2 A_{x+y}^2 + A_i A_{x+y}^3 + A_{x+y}^2 B_\beta + A_{x+y}^2 B_{z(x+y)} + B_\alpha B_{z(x+y)} + B_{xy}B_{z(x+y)},$$
$$A_{c'}^3 A_i + A_{c'}^2 A_i^2 + A_{c'}A_i^3 + A_i^4 + A_i^3 A_m + A_{c'}^2 B_\alpha + A_i^2 B_\alpha + A_i A_m B_\alpha + A_{c'}^2 B_\beta + A_{c'}A_i B_\beta + B_\beta^2,$$
$$A_{c'}A_i A_{x+y}^2 + A_i^2 A_{x+y}^2 + A_i A_{x+y}^3 + A_{x+y}^2 B_\beta + A_i^2 B_{xy} + A_i A_m B_{xy} + B_\beta B_{xy} + A_{x+y}^2 B_{z(x+y)} + B_\alpha B_{z(x+y)},$$
$$A_{c'}^2 A_i A_{x+y} + A_{c'}A_i^2 A_{x+y} + A_{c'}A_{x+y}B_\alpha + A_{c'}A_{x+y}B_\beta + A_i^2 B_{xy} + A_i A_m B_{xy} + A_i^2 B_{z(x+y)} + A_i A_{x+y}B_{z(x+y)}$$
$$+ B_\beta B_{z(x+y)},$$
$$A_{c'}^2 A_{x+y}^2 + A_{c'}A_i A_{x+y}^2 + A_i^2 A_{x+y}^2 + A_{x+y}^4 + A_{c'}A_{x+y}B_\alpha + A_{x+y}^2 B_\beta + A_i^2 B_{xy} + A_i A_m B_{xy} + A_i A_{x+y}B_{z(x+y)}$$
$$+ B_{z(x+y)}^2. \tag{534c}$$

We have the following table regarding IWPs and group cohomology at degree 3.

| Wyckoff position | Little group Intl. | Little group Schönflies | Coordinates | LSM anomaly class | Topo. inv. |
|---|---|---|---|---|---|
| 2a | $mmm$ | $D_{2h}$ | $(0,0,0)$, $(0,0,1/2)$ | $(A_{c'} + A_m + A_i)(A_{c'}A_{x+y} + A_{x+y}^2 + B_\alpha + B_{xy})$ | $\varphi_2[C_2, C_2']$ |
| 2b | $mmm$ | $D_{2h}$ | $(1/2,1/2,0)$, $(1/2,1/2,1/2)$ | $(A_{c'} + A_m + A_i)B_{xy}$ | $\varphi_2[T_1 T_2 C_2, T_1 C_2']$ |
| 2c | $mmm$ | $D_{2h}$ | $(0,1/2,0)$, $(1/2,0,1/2)$ | $A_{x+y}(A_{c'}^2 + A_i^2 + A_{x+y}^2 + B_\beta + B_{z(x+y)})$ | $\varphi_2[T_2 C_2, C_2']$ |
| 2d | $mmm$ | $D_{2h}$ | $(0,1/2,1/2)$, $(1/2,0,0)$ | $(A_{c'} + A_{x+y})(A_{x+y}^2 + B_{z(x+y)})$ | $\varphi_2[T_1 C_2, T_1 C_2']$ |
| 2e | $\bar{4}m2$ | $D_{2d}$ | $(0,0,1/4)$, $(0,0,3/4)$ | $A_m(B_\alpha + B_{xy})$ | $\varphi_2[C_2, GI]$ |
| 2f | $\bar{4}m2$ | $D_{2d}$ | $(1/2,1/2,1/4)$, $(1/2,1/2,3/4)$ | $A_m B_{xy}$ | $\varphi_2[T_1 T_2 C_2, T_1 T_2 GI]$ |

**No. 132:** $P4_2/mcm$

This group is generated by three translations $T_{1,2,3}$ as given in Eqs. (88), a two-fold rotation $C_2$, a two-fold rotation $C_2'$, a mirror $M$, and an inversion $I$:

$$C_2 \colon (x,y,z) \to (-x,-y,z), \tag{535a}$$
$$C_2' \colon (x,y,z) \to (-x,y,-z+1/2), \tag{535b}$$
$$M \colon (x,y,z) \to (y,x,z), \tag{535c}$$
$$I \colon (x,y,z) \to (-x,-y,-z). \tag{535d}$$

The $\mathbb{Z}_2$ cohomology ring is given by

$$\mathbb{Z}_2[A_i, A_m, A_{c'}, A_{x+y}, B_\alpha, B_\beta, B_{xy}]/\langle \mathscr{R}_2, \mathscr{R}_3, \mathscr{R}_4 \rangle \tag{536}$$

where the relations are

$$\mathcal{R}_2: \quad A_{c'}A_i, \quad A_{c'}A_m, \quad A_m A_{x+y}, \tag{537a}$$

$$\mathcal{R}_3: \quad A_{c'}B_\beta, \quad A_{x+y}(A_i^2 + A_i A_{x+y} + B_\beta), \quad A_{c'}A_{x+y}^2 + A_{x+y}^3 + A_{x+y}B_\alpha + A_{c'}B_{xy}, \quad A_{x+y}(A_{c'}A_{x+y} + A_{x+y}^2 + B_\alpha + B_{xy}), \tag{537b}$$

$$\mathcal{R}_4: \quad A_i^4 + A_i^3 A_m + A_i^2 B_\alpha + A_i A_m B_\beta + B_\beta^2, \quad B_{xy}(B_\alpha + B_{xy}). \tag{537c}$$

We have the following table regarding IWPs and group cohomology at degree 3.

| Wyckoff position | Little group | | Coordinates | LSM anomaly class | Topo. inv. |
|---|---|---|---|---|---|
| | Intl. | Schönflies | | | |
| 2a | $mmm$ | $D_{2h}$ | $(0,0,0)$, $(0,0,1/2)$ | $A_i(A_{x+y}^2 + B_\alpha + B_{xy})$ | $\varphi_2[C_2, MI]$ |
| 2b | $\bar{4}2m$ | $D_{2d}$ | $(0,0,1/4)$, $(0,0,3/4)$ | $(A_{c'} + A_{x+y})(A_{c'}A_{x+y} + A_{x+y}^2 + B_\alpha)$ | $\varphi_2[C_2, C_2']$ |
| 2c | $mmm$ | $D_{2h}$ | $(1/2,1/2,0)$, $(1/2,1/2,1/2)$ | $A_i B_{xy}$ | $\varphi_2[T_1 T_2 C_2, T_1 T_2 MI]$ |
| 2d | $\bar{4}2m$ | $D_{2d}$ | $(1/2,1/2,1/4)$, $(1/2,1/2,3/4)$ | $A_{c'}B_{xy}$ | $\varphi_2[T_1 T_2 C_2, T_1 C_2']$ |
| 4e | 222 | $D_2$ | $(0,1/2,1/4)$, $(1/2,0,3/4)$, $(0,1/2,3/4)$, $(1/2,0,1/4)$ | $A_{c'}A_{x+y}(A_{c'} + A_{x+y})$ | $\varphi_2[T_1 C_2, T_1 C_2']$ |
| 4f | $2/m$ | $C_{2h}$ | $(0,1/2,0)$, $(1/2,0,1/2)$, $(0,1/2,1/2)$, $(1/2,0,0)$ | $A_i A_{x+y}^2$ | $\varphi_1[T_1 I]$ |

**No. 133:** $P4_2/nbc$

This group is generated by three translations $T_{1,2,3}$ as given in Eqs. (88), a two-fold rotation $C_2$, a two-fold rotation $C_2'$, a glide $G$, and an inversion $I$:

$$C_2: (x,y,z) \rightarrow (-x+1/2, -y+1/2, z), \tag{538a}$$

$$C_2': (x,y,z) \rightarrow (-x+1/2, y, -z), \tag{538b}$$

$$G: (x,y,z) \rightarrow (y+1/2, x+1/2, z+1/2), \tag{538c}$$

$$I: (x,y,z) \rightarrow (-x, -y, -z). \tag{538d}$$

The $\mathbb{Z}_2$ cohomology ring is given by

$$\mathbb{Z}_2[A_i, A_m, A_{c'}, B_\alpha, B_{\beta 1}, B_{\beta 2}, C_\gamma]/\langle \mathcal{R}_2, \mathcal{R}_3, \mathcal{R}_4, \mathcal{R}_5, \mathcal{R}_6 \rangle \tag{539}$$

where the relations are

$$\mathcal{R}_2: \quad A_{c'}A_i, \quad A_{c'}A_m, \quad A_m(A_i + A_m), \tag{540a}$$

$$\mathcal{R}_3: \quad A_i B_{\beta 1}, \quad A_i^3 + A_i^2 A_m + A_i B_\alpha + A_m B_\alpha + A_m B_{\beta 1}, \quad A_i B_{\beta 2}, \quad A_m B_{\beta 2}, \tag{540b}$$

$$\mathcal{R}_4: \quad (A_i + A_m)C_\gamma, \quad B_{\beta 1}(B_\alpha + B_{\beta 1}), \quad A_{c'}^2 B_\alpha + A_{c'}^2 B_{\beta 2} + B_\alpha B_{\beta 2} + B_{\beta 1}B_{\beta 2} + A_{c'}C_\gamma, \quad A_{c'}^2 B_\alpha + A_{c'}^2 B_{\beta 2} + B_{\beta 2}^2, \tag{540c}$$

$$\mathcal{R}_5: \quad A_i^5 + A_i^4 A_m + A_i^3 B_\alpha + A_i B_\alpha^2 + A_m B_\alpha^2 + A_{c'}B_\alpha B_{\beta 1} + A_i^3 B_{\beta 2} + A_{c'}B_\alpha B_{\beta 2} + A_{c'}^2 C_\gamma + B_{\beta 1}C_\gamma,$$
$$\quad A_{c'}B_\alpha^2 + A_{c'}B_\alpha B_{\beta 1} + A_{c'}B_\alpha B_{\beta 2} + A_{c'}^2 C_\gamma + B_{\beta 2}C_\gamma, \tag{540d}$$

$$\mathcal{R}_6: \quad A_i^5 A_m + A_{c'}^4 B_\alpha + A_i^4 B_\alpha + B_\alpha^3 + B_\alpha^2 B_{\beta 1} + A_{c'}^4 B_{\beta 2} + A_{c'}^2 B_\alpha B_{\beta 2} + A_{c'}B_\alpha C_\gamma + A_i B_\alpha C_\gamma + C_\gamma^2. \tag{540e}$$

We have the following table regarding IWPs and group cohomology at degree 3.

| Wyckoff position | Little group Intl. | Little group Schönflies | Coordinates | LSM anomaly class | Topo. inv. |
|---|---|---|---|---|---|
| 4a | 222 | $D_2$ | $(1/4,1/4,0)$, $(1/4,1/4,1/2)$, $(3/4,3/4,0)$, $(3/4,3/4,1/2)$ | $A_{c'}(B_\alpha + B_{\beta 1})$ | $\varphi_2[C_2, C_2']$ |
| 4b | 222 | $D_2$ | $(3/4,1/4,0)$, $(1/4,3/4,1/2)$, $(1/4,3/4,0)$, $(3/4,1/4,1/2)$ | $A_i^3 + A_i^2 A_m + A_i B_\alpha + A_m B_\alpha + A_{c'} B_{\beta 1}$ | $\varphi_2[T_1 C_2, T_1 C_2']$ |
| 4c | 222 | $D_2$ | $(1/4,1/4,1/4)$, $(1/4,1/4,3/4)$, $(3/4,3/4,3/4)$, $(3/4,3/4,1/4)$ | $A_i(A_i^2 + A_i A_m + B_\alpha)$ | $\varphi_2[C_2, GI]$ |
| 4d | $\bar{4}$ | $S_4$ | $(3/4,1/4,3/4)$, $(1/4,3/4,1/4)$, $(3/4,1/4,1/4)$, $(1/4,3/4,3/4)$ | $(A_i + A_m)(A_i^2 + B_\alpha)$ | $\varphi_2[C_2 C_2' G, T_2^{-1} C_2]$ |
| 8e | $\bar{1}$ | $C_i$ | $(0,0,0)$, $(1/2,1/2,0)$, $(1/2,0,1/2)$, $(0,1/2,1/2)$, $(1/2,0,0)$, $(0,1/2,0)$, $(0,0,1/2)$, $(1/2,1/2,1/2)$ | $A_i^2(A_i + A_m)$ | $\varphi_1[I]$ |

**No. 134:** $P4_2/nnm$

This group is generated by three translations $T_{1,2,3}$ as given in Eqs. (88), a two-fold rotation $C_2$, a two-fold rotation $C_2'$, a glide $G$, and an inversion $I$:

$$C_2 : (x,y,z) \rightarrow (-x+1/2, -y+1/2, z), \tag{541a}$$
$$C_2' : (x,y,z) \rightarrow (-x+1/2, y, -z+1/2), \tag{541b}$$
$$G : (x,y,z) \rightarrow (y+1/2, x+1/2, z), \tag{541c}$$
$$I : (x,y,z) \rightarrow (-x, -y, -z). \tag{541d}$$

The $\mathbb{Z}_2$ cohomology ring is given by

$$\mathbb{Z}_2[A_i, A_m, A_{c'}, A_{x+y+z}, B_\alpha]/\langle \mathcal{R}_2, \mathcal{R}_3, \mathcal{R}_4 \rangle \tag{542}$$

where the relations are

$$\mathcal{R}_2: \quad A_{c'} A_i, \quad A_{c'} A_m, \tag{543a}$$
$$\mathcal{R}_3: \quad (A_i + A_m)(A_m + A_{x+y+z})(A_i + A_m + A_{x+y+z}), \quad A_i(A_i^2 + A_m^2 + A_i A_{x+y+z} + A_{x+y+z}^2 + B_\alpha), \tag{543b}$$
$$\mathcal{R}_4: \quad A_i^3 A_m + A_i A_m^3 + A_i^3 A_{x+y+z} + A_{c'}^2 A_{x+y+z}^2 + A_m^2 A_{x+y+z}^2 + A_i A_{x+y+z}^3 + A_{x+y+z}^4 + A_m^2 B_\alpha$$
$$+ A_{c'} A_{x+y+z} B_\alpha + A_{x+y+z}^2 B_\alpha. \tag{543c}$$

We have the following table regarding IWPs and group cohomology at degree 3.

| Wyckoff position | Little group Intl. | Little group Schönflies | Coordinates | LSM anomaly class | Topo. inv. |
|---|---|---|---|---|---|
| 2a | $\bar{4}2m$ | $D_{2d}$ | $(1/4,3/4,1/4)$, $(3/4,1/4,3/4)$ | $(A_{c'} + A_m + A_{x+y+z})(A_{c'} A_{x+y+z} + A_m A_{x+y+z} + A_{x+y+z}^2 + B_\alpha)$ | $\varphi_2[T_2 C_2, C_2']$ |
| 2b | $\bar{4}2m$ | $D_{2d}$ | $(3/4,1/4,1/4)$, $(1/4,3/4,3/4)$ | $A_m^2 A_{x+y+z} + A_{c'} A_{x+y+z}^2 + A_{x+y+z}^3 + A_m B_\alpha + A_{x+y+z} B_\alpha$ | $\varphi_2[T_1 C_2, T_1 C_2']$ |
| 4c | 222 | $D_2$ | $(1/4,1/4,1/4)$, $(1/4,1/4,3/4)$, $(3/4,3/4,3/4)$, $(3/4,3/4,1/4)$ | $A_{c'} A_{x+y+z}(A_{c'} + A_{x+y+z})$ | $\varphi_2[C_2, C_2']$ |
| 4d | 222 | $D_2$ | $(1/4,1/4,0)$, $(1/4,1/4,1/2)$, $(3/4,3/4,0)$, $(3/4,3/4,1/2)$ | $A_i(A_m + A_{x+y+z})(A_i + A_m + A_{x+y+z})$ | $\varphi_2[C_2, GI]$ |
| 4e | $2/m$ | $C_{2h}$ | $(0,0,1/2)$, $(1/2,1/2,1/2)$, $(1/2,0,0)$, $(0,1/2,0)$ | $A_i(A_i + A_m)(A_i + A_m + A_{x+y+z})$ | $\varphi_1[T_3 I]$ |
| 4f | $2/m$ | $C_{2h}$ | $(0,0,0)$, $(1/2,1/2,0)$, $(1/2,0,1/2)$, $(0,1/2,1/2)$ | $A_i(A_i + A_m)(A_m + A_{x+y+z})$ | $\varphi_1[I]$ |

**No. 135:** $P4_2/mbc$

This group is generated by three translations $T_{1,2,3}$ as given in Eqs. (88), a two-fold rotation $C_2$, a two-fold screw $S_2'$, a glide $G$, and an inversion $I$:

$$C_2 \colon (x,y,z) \to (-x,-y,z), \tag{544a}$$
$$S_2' \colon (x,y,z) \to (-x+1/2, y+1/2, -z), \tag{544b}$$
$$G \colon (x,y,z) \to (y+1/2, x+1/2, z+1/2), \tag{544c}$$
$$I \colon (x,y,z) \to (-x,-y,-z). \tag{544d}$$

The $\mathbb{Z}_2$ cohomology ring is given by

$$\mathbb{Z}_2[A_i, A_m, A_{c'}, B_\alpha, B_{\beta 1}, B_{\beta 2}, C_\gamma]/\langle \mathcal{R}_2, \mathcal{R}_3, \mathcal{R}_4, \mathcal{R}_5, \mathcal{R}_6 \rangle \tag{545}$$

where the relations are

$$\mathcal{R}_2 \colon \quad A_{c'}A_m, \quad A_m(A_i + A_m), \quad A_{c'}^2, \tag{546a}$$
$$\mathcal{R}_3 \colon \quad A_{c'}B_\alpha + A_m B_{\beta 1}, \quad A_{c'}(A_i^2 + B_\alpha + B_{\beta 1}), \quad A_m B_{\beta 2}, \quad A_{c'}(A_i^2 + B_{\beta 2}), \tag{546b}$$
$$\mathcal{R}_4 \colon \quad A_{c'}(A_i^3 + C_\gamma), \quad A_i^4 + A_i^3 A_m + A_i^2 B_\alpha + A_i A_m B_\alpha + B_\alpha B_{\beta 1} + B_{\beta 1}^2,$$
$$A_{c'}A_i^3 + A_i^4 + A_i^3 A_m + A_i^2 B_\alpha + A_i A_m B_\alpha + A_i^2 B_{\beta 2} + B_\alpha B_{\beta 2} + B_{\beta 1}B_{\beta 2} + A_i C_\gamma + A_m C_\gamma,$$
$$A_i^4 + A_i^3 A_m + A_i^2 B_\alpha + A_i A_m B_\alpha + B_{\beta 2}^2, \tag{546c}$$
$$\mathcal{R}_5 \colon \quad A_i^5 + A_i^4 A_m + A_i^3 B_\alpha + A_i^2 A_m B_\alpha + A_{c'}B_\alpha^2 + A_i^3 B_{\beta 1} + A_i B_\alpha B_{\beta 1} + A_i^2 C_\gamma + A_i A_m C_\gamma + B_{\beta 1}C_\gamma,$$
$$A_{c'}A_i^4 + A_i^5 + A_i^4 A_m + A_{c'}B_\alpha^2 + A_i B_\alpha^2 + A_m B_\alpha^2 + A_i^3 B_{\beta 1} + A_i B_\alpha B_{\beta 1} + A_i^3 B_{\beta 2} + A_i B_\alpha B_{\beta 2} + B_{\beta 2}C_\gamma, \tag{546d}$$
$$\mathcal{R}_6 \colon \quad A_i^6 + A_i^4 B_\alpha + A_i^2 B_\alpha^2 + A_i A_m B_\alpha^2 + B_\alpha^3 + A_i^2 B_\alpha B_{\beta 1} + B_\alpha^2 B_{\beta 1} + A_m B_\alpha C_\gamma + C_\gamma^2. \tag{546e}$$

We have the following table regarding IWPs and group cohomology at degree 3.

| Wyckoff position | Little group Intl. | Little group Schönflies | Coordinates | LSM anomaly class | Topo. inv. |
|---|---|---|---|---|---|
| 4a | $2/m$ | $C_{2h}$ | $(0,0,0),\ (0,0,1/2),$ $(1/2,1/2,0),\ (1/2,1/2,1/2)$ | $A_i(A_i^2 + A_i A_m + B_{\beta 1})$ | $\varphi_1[I]$ |
| 4b | $\bar{4}$ | $S_4$ | $(0,0,1/4),\ (0,0,3/4),$ $(1/2,1/2,3/4),\ (1/2,1/2,1/4)$ | $A_{c'}B_\alpha$ | $\varphi_2[T_2^{-1}S_2'G, C_2]$ |
| 4c | $2/m$ | $C_{2h}$ | $(0,1/2,0),\ (1/2,0,1/2),$ $(1/2,0,0),\ (0,1/2,1/2)$ | $A_i^3 + A_i^2 A_m + A_{c'}B_\alpha + A_i B_\alpha + A_m B_\alpha + A_i B_{\beta 1}$ | $\varphi_1[T_1 I]$ |
| 4d | $222$ | $D_2$ | $(0,1/2,1/4),\ (1/2,0,3/4),$ $(0,1/2,3/4),\ (1/2,0,1/4)$ | $(A_{c'} + A_m)B_\alpha$ | $\varphi_2[T_1 C_2, GI]$ |

**No. 136:** $P4_2/mnm$

This group is generated by three translations $T_{1,2,3}$ as given in Eqs. (88), a two-fold rotation $C_2$, a two-fold screw $S_2'$, a mirror $M$, and an inversion $I$:

$$C_2 \colon (x,y,z) \to (-x,-y,z), \tag{547a}$$
$$S_2' \colon (x,y,z) \to (-x+1/2, y+1/2, -z+1/2), \tag{547b}$$
$$M \colon (x,y,z) \to (y,x,z), \tag{547c}$$
$$I \colon (x,y,z) \to (-x,-y,-z). \tag{547d}$$

The $\mathbb{Z}_2$ cohomology ring is given by

$$\mathbb{Z}_2[A_i, A_m, A_{c'}, B_\alpha, B_{\beta 1}, B_{\beta 2}, B_{\beta 3}, C_{\gamma 1}, C_{\gamma 2}, D_\delta]/\langle \mathcal{R}_2, \mathcal{R}_3, \mathcal{R}_4, \mathcal{R}_5, \mathcal{R}_6 \rangle \tag{548}$$

where the relations are

$$\mathscr{R}_2: \quad A_{c'}A_i, \quad A_{c'}A_m, \quad A_{c'}^2, \tag{549a}$$

$$\mathscr{R}_3: \quad A_{c'}B_\alpha + A_mB_{\beta1}, \quad A_{c'}(B_\alpha + B_{\beta1}), \quad A_mB_{\beta2}, \quad A_{c'}B_{\beta2}, \quad A_{c'}B_{\beta3}, \tag{549b}$$

$$\mathscr{R}_4: \quad A_{c'}C_{\gamma1}, \quad A_{c'}C_{\gamma2}, \quad A_i^2A_m^2 + A_i^2B_{\beta1} + B_\alpha B_{\beta2} + B_\alpha B_{\beta3} + A_iC_{\gamma1} + A_mC_{\gamma2}, \quad B_{\beta1}(B_\alpha + B_{\beta1}),$$
$$(B_\alpha + B_{\beta1})B_{\beta2}, \quad A_i^2B_{\beta1} + B_\alpha B_{\beta2} + B_{\beta1}B_{\beta3}, \quad A_i^2B_{\beta1} + B_{\beta2}^2, \quad A_i^2B_{\beta1} + A_i^2B_{\beta2} + B_{\beta2}B_{\beta3},$$
$$A_i^4 + A_i^3A_m + A_i^2B_\alpha + A_iA_mB_{\beta3} + B_{\beta3}^2, \tag{549c}$$

$$\mathscr{R}_5: \quad A_{c'}D_\delta, \quad B_{\beta1}C_{\gamma1}, \quad B_{\beta2}C_{\gamma1}, \quad B_{\beta1}C_{\gamma2}, \quad B_{\beta2}C_{\gamma2},$$
$$A_i^4A_m + A_i^3A_m^2 + A_iA_m^4 + A_i^3B_\alpha + A_iA_m^2B_\alpha + A_iB_\alpha^2 + A_i^3B_{\beta1} + A_iB_\alpha B_{\beta1} + B_{\beta3}C_{\gamma1} + A_mD_\delta,$$
$$A_i^5 + A_i^4A_m + A_i^3A_m^2 + A_i^2A_m^3 + A_i^3B_\alpha + A_i^2A_mB_\alpha + A_i^3B_{\beta1} + A_i^3B_{\beta2} + A_i^2A_mB_{\beta3} + A_i^2C_{\gamma1}$$
$$+ A_iA_mC_{\gamma2} + B_{\beta3}C_{\gamma2} + A_iD_\delta, \tag{549d}$$

$$\mathscr{R}_6: \quad A_i^4B_{\beta1} + A_i^2B_\alpha B_{\beta2} + B_{\beta1}D_\delta, \quad A_i^4B_{\beta1} + A_i^4B_{\beta2} + B_{\beta2}D_\delta, \quad C_{\gamma2}(A_iB_\alpha + C_{\gamma2}),$$
$$A_i^4A_m^2 + A_i^3A_m^3 + A_i^2A_m^4 + A_i^4B_\alpha + A_i^3A_mB_\alpha + A_i^2A_m^2B_\alpha + A_i^2B_\alpha^2 + A_i^2B_\alpha B_{\beta1} + A_i^4B_{\beta2} + A_i^4B_{\beta3} + A_i^3A_mB_{\beta3}$$
$$+ A_iA_m^3B_{\beta3} + A_i^3C_{\gamma1} + A_i^2A_mC_{\gamma1} + A_i^3C_{\gamma2} + A_iA_m^2C_{\gamma2} + A_iB_\alpha C_{\gamma2} + A_iA_mD_\delta + B_{\beta3}D_\delta,$$
$$A_i^2A_m^4 + A_iA_m^3B_\alpha + A_i^2B_\alpha^2 + A_iA_mB_\alpha^2 + B_\alpha^3 + A_i^2B_\alpha B_{\beta1} + B_\alpha^2B_{\beta1} + A_mB_\alpha C_{\gamma1} + C_{\gamma1}^2,$$
$$A_i^3A_m^3 + A_i^4B_\alpha + A_i^3A_mB_\alpha + A_i^2A_m^2B_\alpha + A_iA_m^3B_\alpha + A_i^2B_\alpha^2 + A_iA_mB_\alpha^2 + A_i^2B_\alpha B_{\beta1} + A_i^2B_\alpha B_{\beta2} + A_i^2A_mC_{\gamma1}$$
$$+ A_iB_\alpha C_{\gamma1} + C_{\gamma1}C_{\gamma2} + B_\alpha D_\delta, \tag{549e}$$

$$\mathscr{R}_7: \quad A_i^3A_m^4 + A_iA_m^4B_\alpha + A_i^3B_\alpha^2 + A_i^2A_mB_\alpha^2 + A_iA_m^2B_\alpha^2 + A_iB_\alpha^3 + A_i^3B_\alpha B_{\beta1} + A_iB_\alpha^2B_{\beta1} + A_i^2A_m^3B_{\beta3} + A_i^4C_{\gamma1}$$
$$+ A_i^3A_mC_{\gamma1} + A_i^2A_m^2C_{\gamma1} + A_iA_m^3C_{\gamma1} + A_i^2B_\alpha C_{\gamma1} + A_iA_mB_\alpha C_{\gamma1} + A_iA_m^3C_{\gamma2} + A_i^2B_\alpha C_{\gamma2} + A_iA_mB_\alpha C_{\gamma2}$$
$$+ B_\alpha^2C_{\gamma2} + A_mB_\alpha D_\delta + C_{\gamma1}D_\delta,$$
$$A_i^6A_m + A_i^5A_m^2 + A_i^4A_m^3 + A_i^3A_m^4 + A_i^4A_mB_\alpha + A_i^3A_m^2B_\alpha + A_i^3A_m^2B_{\beta3} + A_i^3A_mC_{\gamma1} + A_i^4C_{\gamma2} + A_i^3A_mC_{\gamma2}$$
$$+ A_i^2A_m^2C_{\gamma2} + A_iA_m^3C_{\gamma2} + A_i^2B_\alpha C_{\gamma2} + A_iA_mB_\alpha C_{\gamma2} + A_i^2A_mD_\delta + C_{\gamma2}D_\delta, \tag{549f}$$

$$\mathscr{R}_8: \quad A_i^8 + A_i^5A_m^3 + A_i^4A_m^4 + A_i^2A_m^6 + A_i^5A_mB_\alpha + A_i^2A_m^4B_\alpha + A_i^2B_\alpha^3 + A_i^6B_{\beta1} + A_i^2B_\alpha^2B_{\beta1} + A_i^3A_m^3B_{\beta3}$$
$$+ A_i^3A_m^2C_{\gamma1} + A_i^2A_m^3C_{\gamma2} + A_i^3B_\alpha C_{\gamma2} + A_i^2A_mB_\alpha C_{\gamma2} + A_iB_\alpha^2C_{\gamma2} + A_iA_mB_\alpha D_\delta + D_\delta^2. \tag{549g}$$

We have the following table regarding IWPs and group cohomology at degree 3.

| Wyckoff position | Little group Intl. | Little group Schönflies | Coordinates | LSM anomaly class | Topo. inv. |
|---|---|---|---|---|---|
| 2a | $mmm$ | $D_{2h}$ | $(0,0,0)$, $(1/2,1/2,1/2)$ | $A_iB_\alpha + A_iB_{\beta1} + C_{\gamma2}$ | $\varphi_2[C_2, MI]$ |
| 2b | $mmm$ | $D_{2h}$ | $(0,0,1/2)$, $(1/2,1/2,0)$ | $C_{\gamma2}$ | $\varphi_2[C_2, T_3MI]$ |
| 4c | $2/m$ | $C_{2h}$ | $(0,1/2,0)$, $(0,1/2,1/2)$, $(1/2,0,1/2)$, $(1/2,0,0)$ | $A_iB_{\beta1}$ | $\varphi_1[T_1I]$ |
| 4d | $\overline{4}$ | $S_4$ | $(0,1/2,1/4)$, $(0,1/2,3/4)$, $(1/2,0,1/4)$, $(1/2,0,3/4)$ | $A_{c'}B_\alpha$ | $\varphi_2[S_2'M, T_2C_2]$ |

**No. 137:** $P4_2/nmc$

This group is generated by three translations $T_{1,2,3}$ as given in Eqs. (88), a two-fold rotation $C_2$, a two-fold screw $S_2'$, a glide $G$, and an inversion $I$:

$$C_2: (x,y,z) \to (-x+1/2, -y+1/2, z), \tag{550a}$$

$$S_2': (x,y,z) \to (-x, y+1/2, -z), \tag{550b}$$

$$G: (x,y,z) \to (y, x, z+1/2), \tag{550c}$$

$$I: (x,y,z) \to (-x, -y, -z). \tag{550d}$$

The $\mathbb{Z}_2$ cohomology ring is given by

$$\mathbb{Z}_2[A_i, A_{c'}, A_m, B_\alpha, B_{\beta1}, B_{\beta2}, C_\gamma]/\langle\mathscr{R}_2, \mathscr{R}_3, \mathscr{R}_4, \mathscr{R}_5, \mathscr{R}_6\rangle \tag{551}$$

where the relations are

$$\mathscr{R}_2: \quad A_{c'}A_m, \quad A_{c'}(A_{c'}+A_i), \quad A_m(A_i+A_m), \tag{552a}$$

$$\mathscr{R}_3: \quad A_{c'}A_i^2 + A_i^3 + A_i^2 A_m + A_{c'}B_\alpha + A_i B_\alpha + A_{c'}B_{\beta1} + A_i B_{\beta1}, \quad A_{c'}A_i^2 + A_i^3 + A_i^2 A_m + A_{c'}B_\alpha + A_i B_\alpha + A_m B_{\beta1},$$
$$(A_{c'}+A_i)B_{\beta2}, \quad A_m B_{\beta2}, \tag{552b}$$

$$\mathscr{R}_4: \quad (A_{c'}+A_i+A_m)C_\gamma, \quad B_{\beta1}(B_\alpha+B_{\beta1}), \quad B_{\beta1}B_{\beta2}+A_{c'}C_\gamma, \quad B_{\beta2}^2, \tag{552c}$$

$$\mathscr{R}_5: \quad (B_\alpha+B_{\beta1})C_\gamma, \quad B_{\beta2}C_\gamma, \tag{552d}$$

$$\mathscr{R}_6: \quad C_\gamma(A_{c'}B_\alpha+A_i B_\alpha+C_\gamma). \tag{552e}$$

We have the following table regarding IWPs and group cohomology at degree 3.

| Wyckoff position | Little group Intl. | Little group Schönflies | Coordinates | LSM anomaly class | Topo. inv. |
|---|---|---|---|---|---|
| 2a | $\bar{4}m2$ | $D_{2d}$ | $(3/4,1/4,3/4), (1/4,3/4,1/4)$ | $A_{c'}A_i^2 + A_i^3 + A_i^2 A_m + A_{c'}B_\alpha + A_i B_\alpha + C_\gamma$ | $\varphi_2[T_1 C_2, T_1 T_2 T_3 GI]$ |
| 2b | $\bar{4}m2$ | $D_{2d}$ | $(3/4,1/4,1/4), (1/4,3/4,3/4)$ | $C_\gamma$ | $\varphi_2[T_1 C_2, T_1 T_2 GI]$ |
| 4d | $mm2$ | $C_{2v}$ | $(1/4,1/4,z), (1/4,1/4,z+1/2),$ $(3/4,3/4,-z), (3/4,3/4,-z+1/2)$ | $(A_{c'}+A_i+A_m)(A_i^2+B_\alpha)$ | $\varphi_2[S_2'GI, C_2]$ |
| 8e | $\bar{1}$ | $C_i$ | $(0,0,0), (1/2,1/2,0),$ $(1/2,0,1/2), (0,1/2,1/2),$ $(0,1/2,0), (1/2,0,0),$ $(1/2,1/2,1/2), (0,0,1/2)$ | $A_i^2(A_{c'}+A_i+A_m)$ | $\varphi_1[I]$ |

## No. 138: $P4_2/ncm$

This group is generated by three translations $T_{1,2,3}$ as given in Eqs. (88), a two-fold rotation $C_2$, a two-fold screw $S_2'$, a mirror $M$, and an inversion $I$:

$$C_2: (x,y,z) \to (-x+1/2, -y+1/2, z), \tag{553a}$$

$$S_2': (x,y,z) \to (-x, y+1/2, -z+1/2), \tag{553b}$$

$$M: (x,y,z) \to (y,x,z), \tag{553c}$$

$$I: (x,y,z) \to (-x,-y,-z). \tag{553d}$$

The $\mathbb{Z}_2$ cohomology ring is given by

$$\mathbb{Z}_2[A_i, A_m, A_{c'}, B_\alpha, B_{\beta1}, B_{\beta2}, B_{\beta3}]/\langle \mathscr{R}_2, \mathscr{R}_3, \mathscr{R}_4 \rangle \tag{554}$$

where the relations are

$$\mathscr{R}_2: \quad A_{c'}A_i, \quad A_{c'}A_m, \quad A_{c'}^2, \tag{555a}$$

$$\mathscr{R}_3: \quad A_{c'}B_{\beta1}, \quad A_i^3 + A_i^2 A_m + A_{c'}B_\alpha + A_i B_\alpha + A_m B_{\beta2}, \quad A_i^3 + A_i^2 A_m + A_{c'}B_\alpha + A_i B_\alpha + A_{c'}B_{\beta2} + A_i B_{\beta2},$$
$$A_i B_{\beta1} + A_m B_{\beta3}, \quad A_{c'}B_{\beta3}, \tag{555b}$$

$$\mathscr{R}_4: \quad A_i^4 + A_i^3 A_m + A_i^2 B_\alpha + A_i A_m B_{\beta1} + B_{\beta1}^2, \quad A_i^2 B_{\beta1} + B_{\beta1}B_{\beta2} + A_i^2 B_{\beta3} + B_\alpha B_{\beta3},$$
$$A_i^4 + A_i^3 A_m + A_i^2 B_\alpha + A_i^2 B_{\beta1} + B_{\beta1}B_{\beta3}, \quad B_{\beta2}(B_\alpha+B_{\beta2}), \quad A_i^2 B_{\beta1} + A_i^2 B_{\beta3} + B_\alpha B_{\beta3} + B_{\beta2}B_{\beta3},$$
$$A_i^4 + A_i^3 A_m + A_i^2 B_\alpha + A_i^2 B_{\beta3} + B_{\beta3}^2. \tag{555c}$$

We have the following table regarding IWPs and group cohomology at degree 3.

| Wyckoff position | Little group Intl. | Little group Schönflies | Coordinates | LSM anomaly class | Topo. inv. |
|---|---|---|---|---|---|
| 4a | 222 | $D_2$ | $(3/4, 1/4, 0)$, $(1/4, 3/4, 1/2)$, $(1/4, 3/4, 0)$, $(3/4, 1/4, 1/2)$ | $A_i B_{\beta 2}$ | $\varphi_2[T_1^{-1}C_2, MI]$ |
| 4b | $\bar{4}$ | $S_4$ | $(3/4, 1/4, 3/4)$, $(1/4, 3/4, 1/4)$, $(1/4, 3/4, 3/4)$, $(3/4, 1/4, 1/4)$ | $A_i^3 + A_i^2 A_m + A_{c'} B_\alpha + A_i B_\alpha + A_i B_{\beta 2}$ | $\varphi_2[S_2'M, T_1^{-1}C_2]$ |
| 4c | $2/m$ | $C_{2h}$ | $(0, 0, 1/2)$, $(1/2, 1/2, 1/2)$, $(1/2, 0, 0)$, $(0, 1/2, 0)$ | $A_i(B_{\beta 1} + B_{\beta 3})$ | $\varphi_1[T_1 I]$ |
| 4d | $2/m$ | $C_{2h}$ | $(0, 0, 0)$, $(1/2, 1/2, 0)$, $(1/2, 0, 1/2)$, $(0, 1/2, 1/2)$ | $A_i(A_i^2 + A_i A_m + B_{\beta 1} + B_{\beta 3})$ | $\varphi_1[I]$ |
| 4e | $mm2$ | $C_{2v}$ | $(1/4, 1/4, z)$, $(1/4, 1/4, z+1/2)$, $(3/4, 3/4, -z+1/2)$, $(3/4, 3/4, -z)$ | $A_i(A_i^2 + A_i A_m + B_\alpha + B_{\beta 2})$ | $\varphi_2[S_2'I, C_2]$ |

### No. 139: $I4/mmm$

This group is generated by three translations $T_{1,2,3}$ as given in Eqs. (91), a two-fold rotation $C_2$, a two-fold rotation $C_2'$, a mirror $M$, and an inversion $I$:

$$C_2 \colon (x, y, z) \to (-x, -y, z), \tag{556a}$$
$$C_2' \colon (x, y, z) \to (-x, y, -z), \tag{556b}$$
$$M \colon (x, y, z) \to (y, x, z), \tag{556c}$$
$$I \colon (x, y, z) \to (-x, -y, -z). \tag{556d}$$

The $\mathbb{Z}_2$ cohomology ring is given by

$$\mathbb{Z}_2[A_i, A_m, A_{c'}, A_{x+y+z}, B_\alpha, B_\beta, B_{z(x+y)}, C_{xyz}]/\langle \mathcal{R}_2, \mathcal{R}_3, \mathcal{R}_4, \mathcal{R}_5, \mathcal{R}_6 \rangle \tag{557}$$

where the relations are

$$\mathcal{R}_2 \colon \quad A_{c'}A_m, \quad A_{c'}A_{x+y+z}, \quad A_{x+y+z}(A_i + A_{x+y+z}), \tag{558a}$$

$$\mathcal{R}_3 \colon \quad A_i^2 A_{x+y+z} + A_i A_m A_{x+y+z} + A_{x+y+z}B_\alpha + A_m B_\beta, \quad A_{x+y+z}(A_i^2 + A_i A_m + B_\alpha + B_\beta),$$
$$A_i^2 A_{x+y+z} + A_i A_m A_{x+y+z} + A_{x+y+z}B_\alpha + A_m B_{z(x+y)}, \quad A_{x+y+z}(A_i^2 + A_i A_m + B_\alpha + B_{z(x+y)}), \tag{558b}$$

$$\mathcal{R}_4 \colon \quad A_{x+y+z}C_{xyz},$$
$$A_i^3 A_{x+y+z} + A_i^2 A_m A_{x+y+z} + A_i A_{x+y+z}B_\alpha + A_{c'}^2 B_\beta + A_{c'}A_i B_\beta + B_\beta^2 + A_{c'}A_i B_{z(x+y)} + A_i^2 B_{z(x+y)} + B_\alpha B_{z(x+y)},$$
$$B_\alpha B_\beta + B_\beta B_{z(x+y)} + A_{c'}C_{xyz}, \quad B_{z(x+y)}(B_\alpha + B_{z(x+y)}), \tag{558c}$$

$$\mathcal{R}_5 \colon \quad (A_{c'}^2 + A_{c'}A_i + B_\beta)C_{xyz}, \quad B_{z(x+y)}C_{xyz}, \tag{558d}$$

$$\mathcal{R}_6 \colon \quad C_{xyz}(A_{c'}B_\alpha + A_i B_\alpha + C_{xyz}). \tag{558e}$$

We have the following table regarding IWPs and group cohomology at degree 3.

| Wyckoff position | Little group Intl. | Little group Schönflies | Coordinates $(0,0,0)+$ $(1/2,1/2,1/2)+$ | LSM anomaly class | Topo. inv. |
|---|---|---|---|---|---|
| 2a | $4/mmm$ | $D_{4h}$ | $(0,0,0)$ | $A_i^2 A_{x+y+z} + A_i A_m A_{x+y+z} + A_{c'}B_\alpha + A_i B_\alpha + A_{c'}B_{z(x+y)} + A_i B_{z(x+y)} + C_{xyz}$ | $\varphi_2[C_2, C_2']$ |
| 2b | $4/mmm$ | $D_{4h}$ | $(0,0,1/2)$ | $C_{xyz}$ | $\varphi_2[C_2, T_1 T_2 C_2']$ |
| 4c | $mmm$ | $D_{2h}$ | $(0,1/2,0)$, $(1/2,0,0)$ | $A_i^2 A_{x+y+z} + A_i A_m A_{x+y+z} + A_{x+y+z}B_\alpha + A_{c'}B_{z(x+y)} + A_i B_{z(x+y)}$ | $\varphi_2[T_1 T_3 C_2, C_2']$ |
| 4d | $\bar{4}m2$ | $D_{2d}$ | $(0,1/2,1/4)$, $(1/2,0,1/4)$ | $A_{x+y+z}(A_i^2 + A_i A_m + B_\alpha)$ | $\varphi_2[T_1 T_3 C_2, T_1 T_2 T_3 MI]$ |
| 8f | $2/m$ | $C_{2h}$ | $(1/4,1/4,1/4)$, $(3/4,3/4,1/4)$, $(3/4,1/4,1/4)$, $(1/4,3/4,1/4)$ | $A_i(A_i + A_m)A_{x+y+z}$ | $\varphi_1[T_1 T_2 T_3 I]$ |

### No. 140: $I4/mcm$

This group is generated by three translations $T_{1,2,3}$ as given in Eqs. (91), a two-fold rotation $C_2$, a two-fold rotation $C_2'$, a glide $G$, and an inversion $I$:

$$C_2 \colon (x, y, z) \to (-x, -y, z), \tag{559a}$$
$$C_2' \colon (x, y, z) \to (-x, y, -z + 1/2), \tag{559b}$$
$$G \colon (x, y, z) \to (y, x, z + 1/2), \tag{559c}$$
$$I \colon (x, y, z) \to (-x, -y, -z). \tag{559d}$$

The $\mathbb{Z}_2$ cohomology ring is given by

$$\mathbb{Z}_2[A_i, A_m, A_{c'}, A_{x+y+z}, B_\alpha, B_{z(x+y)}, D_\delta]/\langle \mathscr{R}_2, \mathscr{R}_3, \mathscr{R}_4, \mathscr{R}_5, \mathscr{R}_6 \rangle \tag{560}$$

where the relations are

$$\mathscr{R}_2: \quad A_{c'}A_m, \quad A_{c'}A_{x+y+z}, \quad A_{c'}A_i + A_iA_m + A_m^2 + A_iA_{x+y+z} + A_{x+y+z}^2, \tag{561a}$$

$$\mathscr{R}_3: \quad A_mA_{x+y+z}^2 + A_{x+y+z}^3 + A_{x+y+z}B_\alpha + A_mB_{z(x+y)}, \quad A_{x+y+z}(A_mA_{x+y+z} + A_{x+y+z}^2 + B_\alpha + B_{z(x+y)}), \tag{561b}$$

$$\mathscr{R}_4: \quad B_{z(x+y)}(B_\alpha + B_{z(x+y)}), \tag{561c}$$

$$\mathscr{R}_5: \quad A_{c'}(B_\alpha B_{z(x+y)} + D_\delta), \quad (A_m + A_{x+y+z})D_\delta, \tag{561d}$$

$$\mathscr{R}_6: \quad (B_\alpha + B_{z(x+y)})D_\delta, \tag{561e}$$

$$\begin{aligned} \mathscr{R}_8: \quad & A_i^4A_m^4 + A_i^3A_m^5 + A_i^6A_mA_{x+y+z} + A_i^4A_m^3A_{x+y+z} + A_i^3A_m^4A_{x+y+z} + A_{c'}A_i^5B_\alpha + A_i^5A_mB_\alpha + A_i^4A_m^2B_\alpha \\ & + A_i^5A_{x+y+z}B_\alpha + A_iA_{x+y+z}B_\alpha^3 + A_i^2B_\alpha^2B_{z(x+y)} + B_\alpha^3B_{z(x+y)} + A_iA_mB_\alpha D_\delta + D_\delta^2. \end{aligned} \tag{561f}$$

We have the following table regarding IWPs and group cohomology at degree 3.

| Wyckoff position | Little group Intl. | Little group Schönflies | Coordinates $(0,0,0)+$ $(1/2,1/2,1/2)+$ | LSM anomaly class | Topo. inv. |
|---|---|---|---|---|---|
| 4a | 422 | $D_4$ | $(0,0,1/4), (0,0,3/4)$ | $(A_{c'}+A_m)(A_mA_{x+y+z}+A_{x+y+z}^2+B_\alpha+B_{z(x+y)})$ | $\varphi_2[C_2,C_2']$ |
| 4b | $\overline{4}2m$ | $D_{2d}$ | $(0,1/2,1/4), (1/2,0,1/4)$ | $A_{c'}B_{z(x+y)}$ | $\varphi_2[T_1T_3C_2,C_2']$ |
| 4c | $4/m$ | $C_{4h}$ | $(0,0,0), (0,0,1/2)$ | $A_iB_\alpha + A_mB_\alpha + A_{x+y+z}B_\alpha + A_iB_{z(x+y)}$ | $\varphi_1[I]$ |
| 4d | $mmm$ | $D_{2h}$ | $(0,1/2,0), (1/2,0,0)$ | $A_iB_{z(x+y)}$ | $\varphi_2[T_1T_3C_2,T_3GI]$ |
| 8e | $2/m$ | $C_{2h}$ | $(1/4,1/4,1/4), (3/4,3/4,1/4),$ $(3/4,1/4,1/4), (1/4,3/4,1/4)$ | $A_i(A_m+A_{x+y+z})A_{x+y+z}$ | $\varphi_1[T_1T_2T_3I]$ |

**No. 141: $I4_1/amd$**

This group is generated by three translations $T_{1,2,3}$ as given in Eqs. (91), a two-fold rotation $C_2$, a two-fold rotation $C_2'$, a glide $G$, and an inversion $I$:

$$C_2: (x,y,z) \to (-x,-y+1/2,z), \tag{562a}$$

$$C_2': (x,y,z) \to (-x+1/2,y,-z+1/2), \tag{562b}$$

$$G: (x,y,z) \to (y+1/4,x+1/4,z-1/4), \tag{562c}$$

$$I: (x,y,z) \to (-x,-y,-z). \tag{562d}$$

The $\mathbb{Z}_2$ cohomology ring is given by

$$\mathbb{Z}_2[A_i, A_m, A_{c'}, B_\alpha, B_{z(x+y)}, C_\gamma]/\langle \mathscr{R}_2, \mathscr{R}_3, \mathscr{R}_4, \mathscr{R}_5, \mathscr{R}_6 \rangle \tag{563}$$

where the relations are

$$\mathscr{R}_2: \quad A_{c'}A_m, \quad A_m(A_i + A_m), \tag{564a}$$

$$\mathscr{R}_3: \quad A_{c'}^2A_i + A_i^3 + A_i^2A_m + A_{c'}B_\alpha + A_iB_\alpha + A_mB_\alpha, \quad A_m(B_\alpha + B_{z(x+y)}), \tag{564b}$$

$$\begin{aligned} \mathscr{R}_4: \quad & (A_{c'} + A_i + A_m)C_\gamma, \\ & A_{c'}^2A_i^2 + A_i^4 + A_i^3A_m + A_iA_mB_\alpha + B_\alpha^2 + A_{c'}A_iB_{z(x+y)} + A_i^2B_{z(x+y)} + B_\alpha B_{z(x+y)} + A_iC_\gamma + A_mC_\gamma, \\ & A_{c'}^2A_i^2 + A_i^4 + A_i^3A_m + A_iA_mB_\alpha + B_\alpha^2 + A_{c'}A_iB_{z(x+y)} + A_i^2B_{z(x+y)} + B_{z(x+y)}^2, \end{aligned} \tag{564c}$$

$$\mathscr{R}_5: \quad (B_\alpha + B_{z(x+y)})C_\gamma, \tag{564d}$$

$$\mathscr{R}_6: \quad C_\gamma(A_mB_\alpha + C_\gamma). \tag{564e}$$

We have the following table regarding IWPs and group cohomology at degree 3.

| Wyckoff position | Little group Intl. | Schönflies | Coordinates $(0,0,0)+ \ (1/2,1/2,1/2)+$ | LSM anomaly class | Topo. inv. |
|---|---|---|---|---|---|
| 4a | $\bar{4}m2$ | $D_{2d}$ | $(0,3/4,1/8), (1/2,3/4,3/8)$ | $A_m B_\alpha + C_\gamma$ | $\varphi_2[C_2, GI]$ |
| 4b | $\bar{4}m2$ | $D_{2d}$ | $(0,1/4,3/8), (0,3/4,5/8)$ | $C_\gamma$ | $\varphi_2[T_1 T_3 C_2, T_3 GI]$ |
| 8c | $2m$ | $C_{2h}$ | $(0,0,0), (1/2,0,1/2),$ $(1/4,3/4,1/4), (1/4,1/4,3/4)$ | $A_m B_\alpha + A_{c'} B_{z(x+y)} + A_i B_{z(x+y)}$ | $\varphi_1[I]$ |
| 8d | $2m$ | $C_{2h}$ | $(0,0,1/2), (1/2,0,0),$ $(1/4,3/4,3/4), (1/4,1/4,1/4)$ | $A_{c'}^2 A_i + A_i^3 + A_i^2 A_m + A_m B_\alpha + A_{c'} B_{z(x+y)} + A_i B_{z(x+y)}$ | $\varphi_1[T_1 T_2 I]$ |

**No. 142:** $I4_1/acd$

This group is generated by three translations $T_{1,2,3}$ as given in Eqs. (91), a two-fold rotation $C_2$, a two-fold rotation $C_2'$, a glide $G$, and an inversion $I$:

$$C_2: (x,y,z) \rightarrow (-x, -y+1/2, z), \tag{565a}$$
$$C_2': (x,y,z) \rightarrow (-x+1/2, y, -z), \tag{565b}$$
$$G: (x,y,z) \rightarrow (y+1/4, x+1/4, z+1/4), \tag{565c}$$
$$I: (x,y,z) \rightarrow (-x, -y, -z). \tag{565d}$$

The $\mathbb{Z}_2$ cohomology ring is given by

$$\mathbb{Z}_2[A_i, A_m, A_{c'}, B_\alpha, D_\gamma]/\langle \mathscr{R}_2, \mathscr{R}_3, \mathscr{R}_4, \mathscr{R}_5, \mathscr{R}_8 \rangle \tag{566}$$

where the relations are

$$\mathscr{R}_2: \quad A_{c'} A_m, \quad A_m(A_i + A_m), \tag{567a}$$
$$\mathscr{R}_3: \quad A_{c'} A_i^2, \quad A_i^3 + A_i^2 A_m + A_{c'} B_\alpha + A_i B_\alpha + A_m B_\alpha, \tag{567b}$$
$$\mathscr{R}_4: \quad A_i(A_i + A_m)(A_i^2 + B_\alpha), \tag{567c}$$
$$\mathscr{R}_5: \quad (A_i + A_m)D_\gamma, \quad A_{c'} D_\gamma, \tag{567d}$$
$$\mathscr{R}_8: \quad A_i^8 + A_i^7 A_m + A_i^2 B_\alpha^3 + A_i^2 B_\alpha D_\gamma + D_\gamma^2. \tag{567e}$$

We have the following table regarding IWPs and group cohomology at degree 3.

| Wyckoff position | Little group Intl. | Schönflies | Coordinates $(0,0,0)+ \ (1/2,1/2,1/2)+$ | LSM anomaly class | Topo. inv. |
|---|---|---|---|---|---|
| 8a | $\bar{4}$ | $S_4$ | $(0,1/4,3/8), (0,3/4,5/8),$ $(1/2,1/4,5/8), (1/2,3/4,3/8)$ | $(A_i + A_m)(A_i^2 + B_\alpha)$ | $\varphi_2[C_2'G, C_2]$ |
| 8b | $222$ | $D_2$ | $(0,1/4,1/8), (0,3/4,3/8),$ $(0,3/4,7/8), (0,1/4,5/8)$ | $A_i(A_i^2 + A_i A_m + B_\alpha)$ | $\varphi_2[C_2, GI]$ |
| 16c | $\bar{1}$ | $C_i$ | $(0,0,0), (1/2,0,1/2),$ $(1/4,3/4,1/4), (1/4,1/4,3/4),$ $(1/2,0,0), (0,0,1/2),$ $(1/4,3/4,3/4), (1/4,1/4,1/4)$ | $A_i^2(A_i + A_m)$ | $\varphi_1[I]$ |
| 16e | $2$ | $C_2$ | $(x,0,1/4), (-x+1/2,0,3/4),$ $(1/4,x+3/4,1/2), (1/4,-x+1/4,0),$ $(-x,0,3/4), (x+1/2,0,1/4),$ $(3/4,-x+1/4,1/2), (3/4,x+3/4,0)$ | $A_{c'}^2 A_i$ | $\varphi_2[C_2'I, T_2 C_2 C_2']$ |

**No. 143:** $P3$

This group is generated by three translations $T_{1,2,3}$ as given in Eqs. (88), and a three-fold rotation $C_3$:

$$C_3: (x,y,z) \rightarrow (-y, x-y, z). \tag{568a}$$

The $\mathbb{Z}_2$ cohomology ring is given by

$$\mathbb{Z}_2[A_z, B_{xy}]/\langle \mathscr{R}_2, \mathscr{R}_4 \rangle \tag{569}$$

where the relations are

$$\mathscr{R}_2: \quad A_z^2, \tag{570a}$$

$$\mathscr{R}_4: \quad B_{xy}^2. \tag{570b}$$

We have the following table regarding IWPs and group cohomology at degree 3.

| Wyckoff position | Little group Intl. | Schönflies | Coordinates | LSM anomaly class | Topo. inv. |
|---|---|---|---|---|---|
| 1a | 3 | $C_3$ | $(0,0,z)$ | $A_z B_{xy}$ | $\varphi_3[T_1, T_2, T_3]$ |
| 1b | 3 | $C_3$ | $(1/3, 2/3, z)$ | Same as 1a | Same as 1a |
| 1c | 3 | $C_3$ | $(2/3, 1/3, z)$ | Same as 1a | Same as 1a |

### No. 144: $P3_1$

This group is generated by three translations $T_{1,2,3}$ as given in Eqs. (88), and a three-fold screw $S_3$:

$$S_3: (x, y, z) \rightarrow (-y, x - y, z + 1/3). \tag{571a}$$

The $\mathbb{Z}_2$ cohomology ring is given by

$$\mathbb{Z}_2[A_z, B_{xy}]/\langle \mathscr{R}_2, \mathscr{R}_4 \rangle \tag{572}$$

where the relations are

$$\mathscr{R}_2: \quad A_z^2, \tag{573a}$$

$$\mathscr{R}_4: \quad B_{xy}^2. \tag{573b}$$

We have the following table regarding IWPs and group cohomology at degree 3.

| Wyckoff position | Little group Intl. | Schönflies | Coordinates | LSM anomaly class | Topo. inv. |
|---|---|---|---|---|---|
| 3a | 1 | $C_1$ | $(x, y, z), (-y, x - y, z + 1/3),$ $(-x + y, -x, z + 2/3)$ | $A_z B_{xy}$ | $\varphi_3[T_1, T_2, T_3]$ |

### No. 145: $P3_2$

This group is generated by three translations $T_{1,2,3}$ as given in Eqs. (88), and a three-fold screw $S_3$:

$$S_3: (x, y, z) \rightarrow (-y, x - y, z + 2/3). \tag{574a}$$

The $\mathbb{Z}_2$ cohomology ring is given by

$$\mathbb{Z}_2[A_z, B_{xy}]/\langle \mathscr{R}_2, \mathscr{R}_4 \rangle \tag{575}$$

where the relations are

$$\mathscr{R}_2: \quad A_z^2, \tag{576a}$$

$$\mathscr{R}_4: \quad B_{xy}^2. \tag{576b}$$

We have the following table regarding IWPs and group cohomology at degree 3.

| Wyckoff position | Little group Intl. | Schönflies | Coordinates | LSM anomaly class | Topo. inv. |
|---|---|---|---|---|---|
| 3a | 1 | $C_1$ | $(x, y, z), (-y, x - y, z + 2/3),$ $(-x + y, -x, z + 1/3)$ | $A_z B_{xy}$ | $\varphi_3[T_1, T_2, T_3]$ |

**No. 146: $R3$**

This group is generated by three translations $T_{1,2,3}$ as given in Eqs. (93), and a three-fold rotation $C_3$:

$$C_3 \colon (x, y, z) \to (-y, x - y, z). \tag{577a}$$

The $\mathbb{Z}_2$ cohomology ring is given by

$$\mathbb{Z}_2[A_z, B_{x(y+z)}]/\langle \mathscr{R}_2, \mathscr{R}_4 \rangle \tag{578}$$

where the relations are

$$\mathscr{R}_2 \colon \quad A_z^2, \tag{579a}$$

$$\mathscr{R}_4 \colon \quad B_{x(y+z)}^2. \tag{579b}$$

We have the following table regarding IWPs and group cohomology at degree 3.

| Wyckoff position | Little group Intl. | Little group Schönflies | Coordinates $(0,0,0)+$ $(2/3,1/3,1/3)+$ $(1/3,2/3,2/3)+$ | LSM anomaly class | Topo. inv. |
|---|---|---|---|---|---|
| 3a | 3 | $C_3$ | $(0, 0, z)$ | $A_z B_{x(y+z)}$ | $\varphi_3[T_1, T_2, T_3]$ |

**No. 147: $P\bar{3}$**

This group is generated by three translations $T_{1,2,3}$ as given in Eqs. (88), a three-fold rotation $C_3$, and an inversion $I$:

$$C_3 \colon (x, y, z) \to (-y, x - y, z), \tag{580a}$$

$$I \colon (x, y, z) \to (-x, -y, -z). \tag{580b}$$

The $\mathbb{Z}_2$ cohomology ring is given by

$$\mathbb{Z}_2[A_i, A_z, B_{xy}]/\langle \mathscr{R}_2, \mathscr{R}_4 \rangle \tag{581}$$

where the relations are

$$\mathscr{R}_2 \colon \quad A_z(A_i + A_z), \tag{582a}$$

$$\mathscr{R}_4 \colon \quad B_{xy}(A_i^2 + B_{xy}). \tag{582b}$$

We have the following table regarding IWPs and group cohomology at degree 3.

| Wyckoff position | Little group Intl. | Little group Schönflies | Coordinates | LSM anomaly class | Topo. inv. |
|---|---|---|---|---|---|
| 1a | $\bar{3}$ | $C_{3i}$ | $(0, 0, 0)$ | $(A_i + A_z)(A_i^2 + B_{xy})$ | $\varphi_1[I]$ |
| 1b | $\bar{3}$ | $C_{3i}$ | $(0, 0, 1/2)$ | $A_z(A_i^2 + B_{xy})$ | $\varphi_1[T_3 I]$ |
| 2d | 3 | $C_3$ | $(1/3, 2/3, z), (2/3, 1/3, -z)$ | N/A | N/A |
| 3e | $\bar{1}$ | $C_i$ | $(1/2, 0, 0), (0, 1/2, 0),$ $(1/2, 1/2, 0)$ | $(A_i + A_z)B_{xy}$ | $\varphi_1[T_1 I]$ |
| 3f | $\bar{1}$ | $C_i$ | $(1/2, 0, 1/2), (0, 1/2, 1/2),$ $(1/2, 1/2, 1/2)$ | $A_z B_{xy}$ | $\varphi_1[T_1 T_3 I]$ |

**No. 148: $R\bar{3}$**

This group is generated by three translations $T_{1,2,3}$ as given in Eqs. (93), a three-fold rotation $C_3$, and an inversion $I$:

$$C_3 \colon (x, y, z) \to (-y, x - y, z), \tag{583a}$$

$$I \colon (x, y, z) \to (-x, -y, -z). \tag{583b}$$

The $\mathbb{Z}_2$ cohomology ring is given by

$$\mathbb{Z}_2[A_i, A_z, B_{x(y+z)}]/\langle \mathscr{R}_2, \mathscr{R}_4 \rangle \tag{584}$$

where the relations are

$$\mathscr{R}_2: \quad A_z(A_i + A_z), \tag{585a}$$

$$\mathscr{R}_4: \quad B_{x(y+z)}(A_i^2 + B_{x(y+z)}). \tag{585b}$$

We have the following table regarding IWPs and group cohomology at degree 3.

| Wyckoff position | Little group Intl. | Little group Schönflies | Coordinates $(0,0,0)+$ $(2/3,1/3,1/3)+$ $(1/3,2/3,2/3)+$ | LSM anomaly class | Topo. inv. |
|---|---|---|---|---|---|
| 3a | $\bar{3}$ | $C_{3i}$ | $(0,0,0)$ | $(A_i + A_z)(A_i^2 + B_{x(y+z)})$ | $\varphi_1[I]$ |
| 3b | $\bar{3}$ | $C_{3i}$ | $(0,0,1/2)$ | $A_z B_{x(y+z)}$ | $\varphi_1[T_1^{-2}T_2^{-1}T_3^3 I]$ |
| 9d | $\bar{1}$ | $C_i$ | $(1/2,0,1/2), (0,1/2,1/2), (1/2,1/2,1/2)$ | $A_z(A_i^2 + B_{x(y+z)})$ | $\varphi_1[T_1^{-1}T_2^{-1}T_3^3 I]$ |
| 9e | $\bar{1}$ | $C_i$ | $(1/2,0,0), (0,1/2,0), (1/2,1/2,0)$ | $(A_i + A_z)B_{x(y+z)}$ | $\varphi_1[T_1 I]$ |

### No. 149: $P312$

This group is generated by three translations $T_{1,2,3}$ as given in Eqs. (88), a three-fold rotation $C_3$, and a two-fold rotation $C_2'$:

$$C_3: (x,y,z) \to (-y, x-y, z), \tag{586a}$$

$$C_2': (x,y,z) \to (-y, -x, -z). \tag{586b}$$

The $\mathbb{Z}_2$ cohomology ring is given by

$$\mathbb{Z}_2[A_{c'}, A_z, B_{xy}]/\langle \mathscr{R}_2, \mathscr{R}_4 \rangle \tag{587}$$

where the relations are

$$\mathscr{R}_2: \quad A_z(A_{c'} + A_z), \tag{588a}$$

$$\mathscr{R}_4: \quad B_{xy}^2. \tag{588b}$$

We have the following table regarding IWPs and group cohomology at degree 3.

| Wyckoff position | Little group Intl. | Little group Schönflies | Coordinates | LSM anomaly class | Topo. inv. |
|---|---|---|---|---|---|
| 1a | 32 | $D_3$ | $(0,0,0)$ | $(A_{c'} + A_z)B_{xy}$ | $\varphi_2[T_1 T_2^{-1}, C_2']$ |
| 1b | 32 | $D_3$ | $(0,0,1/2)$ | $A_z B_{xy}$ | $\varphi_2[T_1 T_2^{-1}, T_3 C_2']$ |
| 1c | 32 | $D_3$ | $(1/3,2/3,0)$ | Same as 1a | Same as 1a |
| 1d | 32 | $D_3$ | $(1/3,2/3,1/2)$ | Same as 1b | Same as 1b |
| 1e | 32 | $D_3$ | $(2/3,1/3,0)$ | Same as 1a | Same as 1a |
| 1f | 32 | $D_3$ | $(2/3,1/3,1/2)$ | Same as 1b | Same as 1b |

### No. 150: $P321$

This group is generated by three translations $T_{1,2,3}$ as given in Eqs. (88), a three-fold rotation $C_3$, and a two-fold rotation $C_2'$:

$$C_3: (x,y,z) \to (-y, x-y, z), \tag{589a}$$

$$C_2': (x,y,z) \to (y, x, -z). \tag{589b}$$

The $\mathbb{Z}_2$ cohomology ring is given by

$$\mathbb{Z}_2[A_{c'}, A_z, B_{xy}]/\langle \mathscr{R}_2, \mathscr{R}_4 \rangle \tag{590}$$

where the relations are

$$\mathcal{R}_2: \quad A_z(A_{c'} + A_z), \tag{591a}$$

$$\mathcal{R}_4: \quad B_{xy}^2. \tag{591b}$$

We have the following table regarding IWPs and group cohomology at degree 3.

| Wyckoff position | Little group Intl. | Little group Schönflies | Coordinates | LSM anomaly class | Topo. inv. |
|---|---|---|---|---|---|
| 1a | 32 | $D_3$ | $(0,0,0)$ | $(A_{c'} + A_z)B_{xy}$ | $\varphi_2[T_1T_2, C_2']$ |
| 1b | 32 | $D_3$ | $(0,0,1/2)$ | $A_z B_{xy}$ | $\varphi_2[T_1T_2, T_3C_2']$ |
| 2d | 3 | $C_3$ | $(1/3, 2/3, z), (2/3, 1/3, -z)$ | N/A | N/A |

### No. 151: $P3_112$

This group is generated by three translations $T_{1,2,3}$ as given in Eqs. (88), a three-fold screw $S_3$, and a two-fold rotation $C_2'$:

$$S_3: (x, y, z) \rightarrow (-y, x - y, z + 1/3), \tag{592a}$$

$$C_2': (x, y, z) \rightarrow (-y, -x, -z + 2/3). \tag{592b}$$

The $\mathbb{Z}_2$ cohomology ring is given by

$$\mathbb{Z}_2[A_{c'}, A_z, B_{xy}]/\langle \mathcal{R}_2, \mathcal{R}_4 \rangle \tag{593}$$

where the relations are

$$\mathcal{R}_2: \quad A_z(A_{c'} + A_z), \tag{594a}$$

$$\mathcal{R}_4: \quad B_{xy}^2. \tag{594b}$$

We have the following table regarding IWPs and group cohomology at degree 3.

| Wyckoff position | Little group Intl. | Little group Schönflies | Coordinates | LSM anomaly class | Topo. inv. |
|---|---|---|---|---|---|
| 3a | 2 | $C_2$ | $(x, -x, 1/3), (x, 2x, 2/3), (-2x, -x, 0)$ | $(A_{c'} + A_z)B_{xy}$ | $\varphi_2[T_1T_2^{-1}, C_2']$ |
| 3b | 2 | $C_2$ | $(x, -x, 5/6), (x, 2x, 1/6), (-2x, -x, 1/2)$ | $A_z B_{xy}$ | $\varphi_2[T_1T_2^{-1}, T_3C_2']$ |

### No. 152: $P3_121$

This group is generated by three translations $T_{1,2,3}$ as given in Eqs. (88), a three-fold screw $S_3$, and a two-fold rotation $C_2'$:

$$S_3: (x, y, z) \rightarrow (-y, x - y, z + 1/3), \tag{595a}$$

$$C_2': (x, y, z) \rightarrow (y, x, -z). \tag{595b}$$

The $\mathbb{Z}_2$ cohomology ring is given by

$$\mathbb{Z}_2[A_{c'}, A_z, B_{xy}]/\langle \mathcal{R}_2, \mathcal{R}_4 \rangle \tag{596}$$

where the relations are

$$\mathcal{R}_2: \quad A_z(A_{c'} + A_z), \tag{597a}$$

$$\mathcal{R}_4: \quad B_{xy}^2. \tag{597b}$$

We have the following table regarding IWPs and group cohomology at degree 3.

| Wyckoff position | Little group Intl. | Little group Schönflies | Coordinates | LSM anomaly class | Topo. inv. |
|---|---|---|---|---|---|
| 3a | 2 | $C_2$ | $(x, 0, 1/3), (0, x, 2/3), (-x, -x, 0)$ | $(A_{c'} + A_z)B_{xy}$ | $\varphi_2[T_1T_2, C_2']$ |
| 3b | 2 | $C_2$ | $(x, 0, 5/6), (0, x, 1/6), (-x, -x, 1/2)$ | $A_z B_{xy}$ | $\varphi_2[T_1T_2, T_3C_2']$ |

**No. 153: $P3_212$**

This group is generated by three translations $T_{1,2,3}$ as given in Eqs. (88), a three-fold screw $S_3$, and a two-fold rotation $C_2'$:

$$S_3 \colon (x,y,z) \to (-y, x-y, z+2/3), \tag{598a}$$

$$C_2' \colon (x,y,z) \to (-y, -x, -z+1/3). \tag{598b}$$

The $\mathbb{Z}_2$ cohomology ring is given by

$$\mathbb{Z}_2[A_{c'}, A_z, B_{xy}]/\langle \mathcal{R}_2, \mathcal{R}_4 \rangle \tag{599}$$

where the relations are

$$\mathcal{R}_2 \colon \quad A_z(A_{c'} + A_z), \tag{600a}$$

$$\mathcal{R}_4 \colon \quad B_{xy}^2. \tag{600b}$$

We have the following table regarding IWPs and group cohomology at degree 3.

| Wyckoff position | Little group Intl. | Little group Schönflies | Coordinates | LSM anomaly class | Topo. inv. |
|---|---|---|---|---|---|
| 3a | 2 | $C_2$ | $(x,-x,2/3)$, $(x,2x,1/3)$, $(-2x,-x,0)$ | $A_z B_{xy}$ | $\varphi_2[T_1 T_2^{-1}, T_3 C_2']$ |
| 3b | 2 | $C_2$ | $(x,-x,1/6)$, $(x,2x,5/6)$, $(-2x,-x,1/2)$ | $(A_{c'} + A_z)B_{xy}$ | $\varphi_2[T_1 T_2^{-1}, C_2']$ |

**No. 154: $P3_221$**

This group is generated by three translations $T_{1,2,3}$ as given in Eqs. (88), a three-fold screw $S_3$, and a two-fold rotation $C_2'$:

$$S_3 \colon (x,y,z) \to (-y, x-y, z+2/3), \tag{601a}$$

$$C_2' \colon (x,y,z) \to (y, x, -z). \tag{601b}$$

The $\mathbb{Z}_2$ cohomology ring is given by

$$\mathbb{Z}_2[A_{c'}, A_z, B_{xy}]/\langle \mathcal{R}_2, \mathcal{R}_4 \rangle \tag{602}$$

where the relations are

$$\mathcal{R}_2 \colon \quad A_z(A_{c'} + A_z), \tag{603a}$$

$$\mathcal{R}_4 \colon \quad B_{xy}^2. \tag{603b}$$

We have the following table regarding IWPs and group cohomology at degree 3.

| Wyckoff position | Little group Intl. | Little group Schönflies | Coordinates | LSM anomaly class | Topo. inv. |
|---|---|---|---|---|---|
| 3a | 2 | $C_2$ | $(x,0,2/3)$, $(0,x,1/3)$, $(-x,-x,0)$ | $(A_{c'} + A_z)B_{xy}$ | $\varphi_2[T_1 T_2, C_2']$ |
| 3b | 2 | $C_2$ | $(x,0,1/6)$, $(0,x,5/6)$, $(-x,-x,1/2)$ | $A_z B_{xy}$ | $\varphi_2[T_1 T_2, T_3 C_2']$ |

**No. 155: $R32$**

This group is generated by three translations $T_{1,2,3}$ as given in Eqs. (93), a three-fold rotation $C_3$, and a two-fold rotation $C_2'$:

$$C_3 \colon (x,y,z) \to (-y, x-y, z), \tag{604a}$$

$$C_2' \colon (x,y,z) \to (y, x, -z). \tag{604b}$$

The $\mathbb{Z}_2$ cohomology ring is given by

$$\mathbb{Z}_2[A_{c'}, A_z, B_{x(y+z)}]/\langle \mathcal{R}_2, \mathcal{R}_4\rangle \tag{605}$$

where the relations are

$$\mathcal{R}_2: \quad A_z(A_{c'} + A_z), \tag{606a}$$

$$\mathcal{R}_4: \quad B_{x(y+z)}^2. \tag{606b}$$

We have the following table regarding IWPs and group cohomology at degree 3.

| Wyckoff position | Little group Intl. | Little group Schönflies | Coordinates $(0,0,0)+$ $(2/3,1/3,1/3)+$ $(1/3,2/3,2/3)+$ | LSM anomaly class | Topo. inv. |
|---|---|---|---|---|---|
| 3a | 32 | $D_3$ | $(0,0,0)$ | $(A_{c'}+A_z)B_{x(y+z)}$ | $\varphi_2[T_1T_2, C_2']$ |
| 3b | 32 | $D_3$ | $(0,0,1/2)$ | $A_z B_{x(y+z)}$ | $\varphi_2[T_1T_2, T_1^{-2}T_2^{-1}T_3^3 C_2']$ |

### No. 156: $P3m1$

This group is generated by three translations $T_{1,2,3}$ as given in Eqs. (88), a three-fold rotation $C_3$, and a mirror $M$:

$$C_3: (x,y,z) \to (-y, x-y, z), \tag{607a}$$

$$M: (x,y,z) \to (-y, -x, z). \tag{607b}$$

The $\mathbb{Z}_2$ cohomology ring is given by

$$\mathbb{Z}_2[A_m, A_z, B_{xy}]/\langle \mathcal{R}_2, \mathcal{R}_4\rangle \tag{608}$$

where the relations are

$$\mathcal{R}_2: \quad A_z^2, \tag{609a}$$

$$\mathcal{R}_4: \quad B_{xy}^2. \tag{609b}$$

We have the following table regarding IWPs and group cohomology at degree 3.

| Wyckoff position | Little group Intl. | Little group Schönflies | Coordinates | LSM anomaly class | Topo. inv. |
|---|---|---|---|---|---|
| 1a | $3m$ | $C_{3v}$ | $(0,0,z)$ | $A_z B_{xy}$ | $\varphi_3[T_1T_2^{-1}, T_3, M]$ |
| 1b | $3m$ | $C_{3v}$ | $(1/3,2/3,z)$ | Same as 1a | Same as 1a |
| 1c | $3m$ | $C_{3v}$ | $(2/3,1/3,z)$ | Same as 1a | Same as 1a |

### No. 157: $P31m$

This group is generated by three translations $T_{1,2,3}$ as given in Eqs. (88), a three-fold rotation $C_3$, and a mirror $M$:

$$C_3: (x,y,z) \to (-y, x-y, z), \tag{610a}$$

$$M: (x,y,z) \to (y, x, z). \tag{610b}$$

The $\mathbb{Z}_2$ cohomology ring is given by

$$\mathbb{Z}_2[A_m, A_z, B_{xy}]/\langle \mathcal{R}_2, \mathcal{R}_4\rangle \tag{611}$$

where the relations are

$$\mathcal{R}_2: \quad A_z^2, \tag{612a}$$

$$\mathcal{R}_4: \quad B_{xy}^2. \tag{612b}$$

We have the following table regarding IWPs and group cohomology at degree 3.

| Wyckoff position | Little group Intl. | Little group Schönflies | Coordinates | LSM anomaly class | Topo. inv. |
|---|---|---|---|---|---|
| 1a | $3m$ | $C_{3v}$ | $(0,0,z)$ | $A_z B_{xy}$ | $\varphi_3[T_1T_2, T_3, M]$ |
| 2b | 3 | $C_3$ | $(1/3,2/3,z), (2/3,1/3,z)$ | N/A | N/A |

**No. 158:** $P3c1$

This group is generated by three translations $T_{1,2,3}$ as given in Eqs. (88), a three-fold rotation $C_3$, and a glide $G$:

$$C_3 : (x, y, z) \to (-y, x - y, z), \tag{613a}$$
$$G : (x, y, z) \to (-y, -x, z + 1/2). \tag{613b}$$

The $\mathbb{Z}_2$ cohomology ring is given by

$$\mathbb{Z}_2[A_m, B_{xy}]/\langle \mathscr{R}_2, \mathscr{R}_4 \rangle \tag{614}$$

where the relations are

$$\mathscr{R}_2 : \quad A_m^2, \tag{615a}$$
$$\mathscr{R}_4 : \quad B_{xy}^2. \tag{615b}$$

We have the following table regarding IWPs and group cohomology at degree 3.

| Wyckoff position | Little group Intl. | Little group Schönflies | Coordinates | LSM anomaly class | Topo. inv. |
|---|---|---|---|---|---|
| 2a | 3 | $C_3$ | $(0,0,z)$, $(0,0,z+1/2)$ | $A_m B_{xy}$ | $\widehat{\varphi_3}[T_1, T_2, G]$ |
| 2b | 3 | $C_3$ | $(1/3, 2/3, z)$, $(1/3, 2/3, z+1/2)$ | Same as 2a | Same as 2a |
| 2c | 3 | $C_3$ | $(2/3, 1/3, z)$, $(2/3, 1/3, z+1/2)$ | Same as 2a | Same as 2a |

Here the topological invariant $\widehat{\varphi_3}[T_1, T_2, G]$ can be chosen to be the same as that of group No.9 $Cc$, given by Eq. (164).

**No. 159:** $P31c$

This group is generated by three translations $T_{1,2,3}$ as given in Eqs. (88), a three-fold rotation $C_3$, and a glide $G$:

$$C_3 : (x, y, z) \to (-y, x - y, z), \tag{616a}$$
$$G : (x, y, z) \to (y, x, z + 1/2). \tag{616b}$$

The $\mathbb{Z}_2$ cohomology ring is given by

$$\mathbb{Z}_2[A_m, B_{xy}]/\langle \mathscr{R}_2, \mathscr{R}_4 \rangle \tag{617}$$

where the relations are

$$\mathscr{R}_2 : \quad A_m^2, \tag{618a}$$
$$\mathscr{R}_4 : \quad B_{xy}^2. \tag{618b}$$

We have the following table regarding IWPs and group cohomology at degree 3.

| Wyckoff position | Little group Intl. | Little group Schönflies | Coordinates | LSM anomaly class | Topo. inv. |
|---|---|---|---|---|---|
| 2a | 3 | $C_3$ | $(0,0,z)$, $(0,0,z+1/2)$ | $A_m B_{xy}$ | $\widehat{\varphi_3}[T_1, T_2^{-1}, G]$ |
| 2b | 3 | $C_3$ | $(1/3, 2/3, z)$, $(2/3, 1/3, z+1/2)$ | Same as 2a | Same as 2a |

Here the topological invariant $\widehat{\varphi_3}[T_1, T_2^{-1}, G]$ can be chosen to be the same as that of group No.9 $Cc$, given by Eq. (164).

**No. 160:** $R3m$

This group is generated by three translations $T_{1,2,3}$ as given in Eqs. (93), a three-fold rotation $C_3$, and a mirror $M$:

$$C_3 : (x, y, z) \to (-y, x - y, z), \tag{619a}$$
$$M : (x, y, z) \to (-y, -x, z). \tag{619b}$$

The $\mathbb{Z}_2$ cohomology ring is given by

$$\mathbb{Z}_2[A_m, A_z, B_{x(y+z)}]/\langle \mathcal{R}_2, \mathcal{R}_4 \rangle \tag{620}$$

where the relations are

$$\mathcal{R}_2: \quad A_z^2, \tag{621a}$$

$$\mathcal{R}_4: \quad B_{x(y+z)}^2. \tag{621b}$$

We have the following table regarding IWPs and group cohomology at degree 3.

| Wyckoff position | Little group Intl. | Little group Schönflies | Coordinates $(0,0,0)+$ $(2/3,1/3,1/3)+$ $(1/3,2/3,2/3)+$ | LSM anomaly class | Topo. inv. |
|---|---|---|---|---|---|
| 3a | $3m$ | $C_{3v}$ | $(0,0,z)$ | $A_z B_{x(y+z)}$ | $\varphi_3[T_1 T_2^{-1}, T_1^{-2} T_2^{-1} T_3^3, M]$ |

### No. 161: $R3c$

This group is generated by three translations $T_{1,2,3}$ as given in Eqs. (93), a three-fold rotation $C_3$, and a glide $G$:

$$C_3: (x,y,z) \to (-y, x-y, z), \tag{622a}$$

$$G: (x,y,z) \to (-y+1/3, -x+2/3, z+1/6). \tag{622b}$$

The $\mathbb{Z}_2$ cohomology ring is given by

$$\mathbb{Z}_2[A_m, B_{x(y+z)}]/\langle \mathcal{R}_2, \mathcal{R}_4 \rangle \tag{623}$$

where the relations are

$$\mathcal{R}_2: \quad A_m^2, \tag{624a}$$

$$\mathcal{R}_4: \quad B_{x(y+z)}^2. \tag{624b}$$

We have the following table regarding IWPs and group cohomology at degree 3.

| Wyckoff position | Little group Intl. | Little group Schönflies | Coordinates $(0,0,0)+$ $(2/3,1/3,1/3)+$ $(1/3,2/3,2/3)+$ | LSM anomaly class | Topo. inv. |
|---|---|---|---|---|---|
| 6a | 3 | $C_3$ | $(0,0,z), (0,0,z+1/2)$ | $A_m B_{x(y+z)}$ | $\widehat{\varphi_3}[T_1, T_2, G]$ |

Here the topological invariant $\widehat{\varphi_3}[T_1, T_2, G]$ can be chosen to be the same as that of group No.9 $Cc$, given by Eq. (164).

### No. 162: $P\bar{3}1m$

This group is generated by three translations $T_{1,2,3}$ as given in Eqs. (88), a three-fold rotation $C_3$, a two-fold rotation $C_2'$, and an inversion $I$:

$$C_3: (x,y,z) \to (-y, x-y, z), \tag{625a}$$

$$C_2': (x,y,z) \to (-y, -x, -z), \tag{625b}$$

$$I: (x,y,z) \to (-x, -y, -z). \tag{625c}$$

The $\mathbb{Z}_2$ cohomology ring is given by

$$\mathbb{Z}_2[A_i, A_{c'}, A_z, B_{xy}]/\langle \mathcal{R}_2, \mathcal{R}_4 \rangle \tag{626}$$

where the relations are

$$\mathcal{R}_2: \quad A_z(A_{c'} + A_i + A_z), \tag{627a}$$

$$\mathcal{R}_4: \quad B_{xy}(A_{c'} A_i + A_i^2 + B_{xy}). \tag{627b}$$

We have the following table regarding IWPs and group cohomology at degree 3.

| Wyckoff position | Little group Intl. | Little group Schönflies | Coordinates | LSM anomaly class | Topo. inv. |
|---|---|---|---|---|---|
| 1a | $\bar{3}m$ | $D_{3d}$ | $(0,0,0)$ | $(A_{c'} + A_i + A_z)(A_{c'}A_i + A_i^2 + B_{xy})$ | $\varphi_1[I]$ |
| 1b | $\bar{3}m$ | $D_{3d}$ | $(0,0,1/2)$ | $A_z(A_{c'}A_i + A_i^2 + B_{xy})$ | $\varphi_1[T_3 I]$ |
| 2c | $32$ | $D_3$ | $(1/3,2/3,0)$, $(2/3,1/3,0)$ | N/A | N/A |
| 2d | $32$ | $D_3$ | $(1/3,2/3,1/2)$, $(2/3,1/3,1/2)$ | N/A | N/A |
| 3f | $2/m$ | $C_{2h}$ | $(1/2,0,0)$, $(0,1/2,0)$, $(1/2,1/2,0)$ | $(A_{c'} + A_i + A_z)B_{xy}$ | $\varphi_1[T_1 I]$ |
| 3g | $2/m$ | $C_{2h}$ | $(1/2,0,1/2)$, $(0,1/2,1/2)$, $(1/2,1/2,1/2)$ | $A_z B_{xy}$ | $\varphi_1[T_1 T_3 I]$ |

## No. 163: $P\bar{3}1c$

This group is generated by three translations $T_{1,2,3}$ as given in Eqs. (88), a three-fold rotation $C_3$, a two-fold rotation $C_2'$, and an inversion $I$:

$$C_3 : (x,y,z) \to (-y, x-y, z), \tag{628a}$$
$$C_2' : (x,y,z) \to (-y, -x, -z+1/2), \tag{628b}$$
$$I : (x,y,z) \to (-x, -y, -z). \tag{628c}$$

The $\mathbb{Z}_2$ cohomology ring is given by

$$\mathbb{Z}_2[A_i, A_{c'}, B_{xy}]/\langle \mathscr{R}_2, \mathscr{R}_4 \rangle \tag{629}$$

where the relations are

$$\mathscr{R}_2 : \quad A_{c'}A_i, \tag{630a}$$
$$\mathscr{R}_4 : \quad B_{xy}(A_i^2 + B_{xy}). \tag{630b}$$

We have the following table regarding IWPs and group cohomology at degree 3.

| Wyckoff position | Little group Intl. | Little group Schönflies | Coordinates | LSM anomaly class | Topo. inv. |
|---|---|---|---|---|---|
| 2a | $32$ | $D_3$ | $(0,0,1/4)$, $(0,0,3/4)$ | $A_{c'}B_{xy}$ | $\varphi_2[T_1 T_2^{-1}, C_2']$ |
| 2b | $\bar{3}$ | $C_{3i}$ | $(0,0,0)$, $(0,0,1/2)$ | $A_i(A_i^2 + B_{xy})$ | $\varphi_1[I]$ |
| 2c | $32$ | $D_3$ | $(1/3,2/3,1/4)$, $(2/3,1/3,3/4)$ | Same as 2a | Same as 2a |
| 2d | $32$ | $D_3$ | $(2/3,1/3,1/4)$, $(1/3,2/3,3/4)$ | Same as 2b | Same as 2b |
| 6g | $\bar{1}$ | $C_i$ | $(1/2,0,0)$, $(0,1/2,0)$, $(1/2,1/2,0)$, $(0,1/2,1/2)$, $(1/2,0,1/2)$, $(1/2,1/2,1/2)$ | $A_i B_{xy}$ | $\varphi_1[T_1 I]$ |

## No. 164: $P\bar{3}m1$

This group is generated by three translations $T_{1,2,3}$ as given in Eqs. (88), a three-fold rotation $C_3$, a two-fold rotation $C_2'$, and an inversion $I$:

$$C_3 : (x,y,z) \to (-y, x-y, z), \tag{631a}$$
$$C_2' : (x,y,z) \to (y, x, -z), \tag{631b}$$
$$I : (x,y,z) \to (-x, -y, -z). \tag{631c}$$

The $\mathbb{Z}_2$ cohomology ring is given by

$$\mathbb{Z}_2[A_i, A_{c'}, A_z, B_{xy}]/\langle \mathscr{R}_2, \mathscr{R}_4 \rangle \tag{632}$$

where the relations are

$$\mathscr{R}_2 : \quad A_z(A_{c'} + A_i + A_z), \tag{633a}$$
$$\mathscr{R}_4 : \quad B_{xy}(A_{c'}A_i + A_i^2 + B_{xy}). \tag{633b}$$

We have the following table regarding IWPs and group cohomology at degree 3.

| Wyckoff position | Little group Intl. | Little group Schönflies | Coordinates | LSM anomaly class | Topo. inv. |
|---|---|---|---|---|---|
| 1a | $\bar{3}m$ | $D_{3d}$ | $(0,0,0)$ | $(A_{c'} + A_i + A_z)(A_{c'}A_i + A_i^2 + B_{xy})$ | $\varphi_1[I]$ |
| 1b | $\bar{3}m$ | $D_{3d}$ | $(0,0,1/2)$ | $A_z(A_{c'}A_i + A_i^2 + B_{xy})$ | $\varphi_1[T_3I]$ |
| 2d | $3m$ | $C_{3v}$ | $(1/3,2/3,z)$, $(2/3,1/3,-z)$ | N/A | N/A |
| 3e | $2m$ | $C_{2h}$ | $(1/2,0,0)$, $(0,1/2,0)$, $(1/2,1/2,0)$ | $(A_{c'} + A_i + A_z)B_{xy}$ | $\varphi_1[T_1I]$ |
| 3f | $2m$ | $C_{2h}$ | $(1/2,0,1/2)$, $(0,1/2,1/2)$, $(1/2,1/2,1/2)$ | $A_z B_{xy}$ | $\varphi_1[T_1T_3I]$ |

## No. 165: $P\bar{3}c1$

This group is generated by three translations $T_{1,2,3}$ as given in Eqs. (88), a three-fold rotation $C_3$, a two-fold rotation $C_2'$, and an inversion $I$:

$$C_3: (x,y,z) \rightarrow (-y, x-y, z), \tag{634a}$$
$$C_2': (x,y,z) \rightarrow (y, x, -z+1/2), \tag{634b}$$
$$I: (x,y,z) \rightarrow (-x, -y, -z). \tag{634c}$$

The $\mathbb{Z}_2$ cohomology ring is given by

$$\mathbb{Z}_2[A_i, A_{c'}, B_{xy}]/\langle \mathcal{R}_2, \mathcal{R}_4 \rangle \tag{635}$$

where the relations are

$$\mathcal{R}_2: \quad A_{c'}A_i, \tag{636a}$$
$$\mathcal{R}_4: \quad B_{xy}(A_i^2 + B_{xy}). \tag{636b}$$

We have the following table regarding IWPs and group cohomology at degree 3.

| Wyckoff position | Little group Intl. | Little group Schönflies | Coordinates | LSM anomaly class | Topo. inv. |
|---|---|---|---|---|---|
| 2a | $32$ | $D_3$ | $(0,0,1/4)$, $(0,0,3/4)$ | $A_{c'}B_{xy}$ | $\varphi_2[T_1T_2, C_2']$ |
| 2b | $\bar{3}$ | $C_{3i}$ | $(0,0,0)$, $(0,0,1/2)$ | $A_i(A_i^2 + B_{xy})$ | $\varphi_1[I]$ |
| 4d | $3$ | $C_3$ | $(1/3,2/3,z)$, $(2/3,1/3,-z+1/2)$, $(2/3,1/3,-z)$, $(1/3,2/3,z+1/2)$ | N/A | N/A |
| 6e | $\bar{1}$ | $C_i$ | $(1/2,0,0)$, $(0,1/2,0)$, $(1/2,1/2,0)$, $(0,1/2,1/2)$, $(1/2,0,1/2)$, $(1/2,1/2,1/2)$ | $A_i B_{xy}$ | $\varphi_1[T_1I]$ |

## No. 166: $R\bar{3}m$

This group is generated by three translations $T_{1,2,3}$ as given in Eqs. (93), a three-fold rotation $C_3$, a two-fold rotation $C_2'$, and an inversion $I$:

$$C_3: (x,y,z) \rightarrow (-y, x-y, z), \tag{637a}$$
$$C_2': (x,y,z) \rightarrow (y, x, -z), \tag{637b}$$
$$I: (x,y,z) \rightarrow (-x, -y, -z). \tag{637c}$$

The $\mathbb{Z}_2$ cohomology ring is given by

$$\mathbb{Z}_2[A_i, A_{c'}, A_z, B_{x(y+z)}]/\langle \mathcal{R}_2, \mathcal{R}_4 \rangle \tag{638}$$

where the relations are

$$\mathcal{R}_2: \quad A_z(A_{c'} + A_i + A_z), \tag{639a}$$
$$\mathcal{R}_4: \quad B_{x(y+z)}(A_{c'}A_i + A_i^2 + B_{x(y+z)}). \tag{639b}$$

We have the following table regarding IWPs and group cohomology at degree 3.

| Wyckoff position | Little group Intl. | Schönflies | Coordinates $(0,0,0)+$  $(2/3,1/3,1/3)+$  $(1/3,2/3,2/3)+$ | LSM anomaly class | Topo. inv. |
|---|---|---|---|---|---|
| 3a | $\bar{3}m$ | $D_{3d}$ | $(0,0,0)$ | $(A_{c'}+A_i+A_z)(A_{c'}A_i+A_i^2+B_{x(y+z)})$ | $\varphi_1[I]$ |
| 3b | $\bar{3}m$ | $D_{3d}$ | $(0,0,1/2)$ | $A_zB_{x(y+z)}$ | $\varphi_1[T_1^{-2}T_2^{-1}T_3^3I]$ |
| 9d | $2/m$ | $C_{2h}$ | $(1/2,0,1/2),(0,1/2,1/2),(1/2,1/2,1/2)$ | $A_z(A_{c'}A_i+A_i^2+B_{x(y+z)})$ | $\varphi_1[T_1^{-1}T_2^{-1}T_3^3I]$ |
| 9e | $*2/m$ | $C_{2h}$ | $(1/2,0,0),(0,1/2,0),(1/2,1/2,0)$ | $(A_{c'}+A_i+A_z)B_{x(y+z)}$ | $\varphi_1[T_1I]$ |

**No. 167: $R\bar{3}c$**

This group is generated by three translations $T_{1,2,3}$ as given in Eqs. (93), a three-fold rotation $C_3$, a two-fold screw $S_2'$, and an inversion $I$:

$$C_3: (x,y,z) \rightarrow (-y, x-y, z), \tag{640a}$$
$$S_2': (x,y,z) \rightarrow (y+1/3, x+2/3, -z+1/6), \tag{640b}$$
$$I: (x,y,z) \rightarrow (-x,-y,-z). \tag{640c}$$

The $\mathbb{Z}_2$ cohomology ring is given by

$$\mathbb{Z}_2[A_i, A_{c'}, B_{x(y+z)}]/\langle \mathcal{R}_2, \mathcal{R}_4 \rangle \tag{641}$$

where the relations are

$$\mathcal{R}_2: \quad A_{c'}A_i, \tag{642a}$$
$$\mathcal{R}_4: \quad B_{x(y+z)}(A_i^2 + B_{x(y+z)}). \tag{642b}$$

We have the following table regarding IWPs and group cohomology at degree 3.

| Wyckoff position | Little group Intl. | Schönflies | Coordinates $(0,0,0)+$  $(2/3,1/3,1/3)+$  $(1/3,2/3,2/3)+$ | LSM anomaly class | Topo. inv. |
|---|---|---|---|---|---|
| 6a | $32$ | $D_3$ | $(0,0,1/4),(0,0,3/4)$ | $A_{c'}B_{x(y+z)}$ | $\varphi_2[T_1T_2, T_1^{-1}T_2^{-1}T_3S_2']$ |
| 6b | $\bar{3}$ | $C_{3i}$ | $(0,0,0),(0,0,1/2)$ | $A_i(A_i^2+B_{x(y+z)})$ | $\varphi_1[I]$ |
| 18d | $\bar{1}$ | $C_i$ | $(1/2,0,0),(0,1/2,0),$ $(1/2,1/2,0),(0,1/2,1/2),$ $(1/2,0,1/2),(1/2,1/2,1/2)$ | $A_iB_{x(y+z)}$ | $\varphi_1[T_1I]$ |

**No. 168: $P6$**

This group is generated by three translations $T_{1,2,3}$ as given in Eqs. (88), a three-fold rotation $C_3$, and a two-fold rotation $C_2$:

$$C_3: (x,y,z) \rightarrow (-y, x-y, z), \tag{643a}$$
$$C_2: (x,y,z) \rightarrow (-x,-y,z). \tag{643b}$$

The $\mathbb{Z}_2$ cohomology ring is given by

$$\mathbb{Z}_2[A_c, A_z, B_{xy}]/\langle \mathcal{R}_2, \mathcal{R}_4 \rangle \tag{644}$$

where the relations are

$$\mathcal{R}_2: \quad A_z^2, \tag{645a}$$
$$\mathcal{R}_4: \quad B_{xy}(A_c^2 + B_{xy}). \tag{645b}$$

We have the following table regarding IWPs and group cohomology at degree 3.

| Wyckoff position | Little group Intl. | Schönflies | Coordinates | LSM anomaly class | Topo. inv. |
|---|---|---|---|---|---|
| 1a | $6$ | $C_6$ | $(0,0,z)$ | $A_z(A_c^2+B_{xy})$ | $\varphi_2[T_3,C_2]$ |
| 2b | $3$ | $C_3$ | $(1/3,2/3,z),(2/3,1/3,z)$ | N/A | N/A |
| 3c | $2$ | $C_2$ | $(1/2,0,z),(0,1/2,z),(1/2,1/2,z)$ | $A_zB_{xy}$ | $\varphi_2[T_3,T_1C_2]$ |

**No. 169:** $P6_1$

This group is generated by three translations $T_{1,2,3}$ as given in Eqs. (88), a three-fold screw $S_3$, and a two-fold screw $S_2$:

$$S_3\colon (x, y, z) \to (-y, x - y, z + 1/3), \tag{646a}$$
$$S_2\colon (x, y, z) \to (-x, -y, z + 1/2). \tag{646b}$$

The $\mathbb{Z}_2$ cohomology ring is given by

$$\mathbb{Z}_2[A_c, B_{xy}]/\langle \mathcal{R}_2, \mathcal{R}_4 \rangle \tag{647}$$

where the relations are

$$\mathcal{R}_2\colon \quad A_c^2, \tag{648a}$$
$$\mathcal{R}_4\colon \quad B_{xy}^2. \tag{648b}$$

We have the following table regarding IWPs and group cohomology at degree 3.

| Wyckoff position | Little group Intl. | Little group Schönflies | Coordinates | LSM anomaly class | Topo. inv. |
|---|---|---|---|---|---|
| 6a | 1 | $C_1$ | $(x, y, z)$, $(-y, x - y, z + 1/3)$, $(-x + y, -x, z + 2/3)$, $(-x, -y, z + 1/2)$, $(y, -x + y, z + 5/6)$, $(x - y, x, z + 1/6)$ | $A_c B_{xy}$ | $\widehat{\varphi_3}[T_1, T_2, S_2]$ |

Here the topological invariant $\widehat{\varphi_3}[T_1, T_2, S_2]$ can be chosen to be the same as that of group No.4 $P2_1$, given by Eq. (148).

**No. 170:** $P6_5$

This group is generated by three translations $T_{1,2,3}$ as given in Eqs. (88), a three-fold screw $S_3$, and a two-fold screw $S_2$:

$$S_3\colon (x, y, z) \to (-y, x - y, z + 2/3), \tag{649a}$$
$$S_2\colon (x, y, z) \to (-x, -y, z + 1/2). \tag{649b}$$

The $\mathbb{Z}_2$ cohomology ring is given by

$$\mathbb{Z}_2[A_c, B_{xy}]/\langle \mathcal{R}_2, \mathcal{R}_4 \rangle \tag{650}$$

where the relations are

$$\mathcal{R}_2\colon \quad A_c^2, \tag{651a}$$
$$\mathcal{R}_4\colon \quad B_{xy}^2. \tag{651b}$$

We have the following table regarding IWPs and group cohomology at degree 3.

| Wyckoff position | Little group Intl. | Little group Schönflies | Coordinates | LSM anomaly class | Topo. inv. |
|---|---|---|---|---|---|
| 6a | 1 | $C_1$ | $(x, y, z)$, $(-y, x - y, z + 2/3)$, $(-x + y, -x, z + 1/3)$, $(-x, -y, z + 1/2)$, $(y, -x + y, z + 1/6)$, $(x - y, x, z + 5/6)$ | $A_c B_{xy}$ | $\widehat{\varphi_3}[T_1, T_2, S_2]$ |

Here the topological invariant $\widehat{\varphi_3}[T_1, T_2, S_2]$ can be chosen to be the same as that of group No.4 $P2_1$, given by Eq. (148).

**No. 171:** $P6_2$

This group is generated by three translations $T_{1,2,3}$ as given in Eqs. (88), a three-fold screw $S_3$, and a two-fold rotation $C_2$:

$$S_3\colon (x, y, z) \to (-y, x - y, z + 2/3), \tag{652a}$$
$$C_2\colon (x, y, z) \to (-x, -y, z). \tag{652b}$$

The $\mathbb{Z}_2$ cohomology ring is given by

$$\mathbb{Z}_2[A_c, A_z, B_{xy}]/\langle \mathcal{R}_2, \mathcal{R}_4 \rangle \tag{653}$$

where the relations are

$$\mathcal{R}_2: \quad A_z^2, \tag{654a}$$

$$\mathcal{R}_4: \quad B_{xy}(A_c^2 + B_{xy}). \tag{654b}$$

We have the following table regarding IWPs and group cohomology at degree 3.

| Wyckoff position | Little group Intl. | Little group Schönflies | Coordinates | LSM anomaly class | Topo. inv. |
|---|---|---|---|---|---|
| 3a | 2 | $C_2$ | $(0,0,z)$, $(0,0,z+2/3)$, $(0,0,z+1/3)$ | $A_z(A_c^2 + B_{xy})$ | $\varphi_2[T_3, C_2]$ |
| 3b | 2 | $C_2$ | $(1/2,1/2,z)$, $(1/2,0,z+2/3)$, $(0,1/2,z+1/3)$ | $A_z B_{xy}$ | $\varphi_2[T_3, T_1 T_2 C_2]$ |

**No. 172:** $P6_4$

This group is generated by three translations $T_{1,2,3}$ as given in Eqs. (88), a three-fold screw $S_3$, and a two-fold rotation $C_2$:

$$S_3: (x,y,z) \rightarrow (-y, x-y, z+1/3), \tag{655a}$$

$$C_2: (x,y,z) \rightarrow (-x, -y, z). \tag{655b}$$

The $\mathbb{Z}_2$ cohomology ring is given by

$$\mathbb{Z}_2[A_c, A_z, B_{xy}]/\langle \mathcal{R}_2, \mathcal{R}_4 \rangle \tag{656}$$

where the relations are

$$\mathcal{R}_2: \quad A_z^2, \tag{657a}$$

$$\mathcal{R}_4: \quad B_{xy}(A_c^2 + B_{xy}). \tag{657b}$$

We have the following table regarding IWPs and group cohomology at degree 3.

| Wyckoff position | Little group Intl. | Little group Schönflies | Coordinates | LSM anomaly class | Topo. inv. |
|---|---|---|---|---|---|
| 3a | 2 | $C_2$ | $(0,0,z)$, $(0,0,z+1/3)$, $(0,0,z+2/3)$ | $A_z(A_c^2 + B_{xy})$ | $\varphi_2[T_3, C_2]$ |
| 3b | 2 | $C_2$ | $(1/2,1/2,z)$, $(1/2,0,z+1/3)$, $(0,1/2,z+2/3)$ | $A_z B_{xy}$ | $\varphi_2[T_3, T_1 T_2 C_2]$ |

**No. 173:** $P6_3$

This group is generated by three translations $T_{1,2,3}$ as given in Eqs. (88), a three-fold rotation $C_3$, and a two-fold screw $S_2$:

$$C_3: (x,y,z) \rightarrow (-y, x-y, z), \tag{658a}$$

$$S_2: (x,y,z) \rightarrow (-x, -y, z+1/2). \tag{658b}$$

The $\mathbb{Z}_2$ cohomology ring is given by

$$\mathbb{Z}_2[A_c, B_{xy}]/\langle \mathcal{R}_2, \mathcal{R}_4 \rangle \tag{659}$$

where the relations are

$$\mathcal{R}_2: \quad A_c^2, \tag{660a}$$

$$\mathcal{R}_4: \quad B_{xy}^2. \tag{660b}$$

We have the following table regarding IWPs and group cohomology at degree 3.

| Wyckoff position | Little group Intl. | Little group Schönflies | Coordinates | LSM anomaly class | Topo. inv. |
|---|---|---|---|---|---|
| 2a | 3 | $C_3$ | $(0,0,z)$, $(0,0,z+1/2)$ | $A_c B_{xy}$ | $\widehat{\varphi_3}[T_1, T_2, S_2]$ |
| 2b | 3 | $C_3$ | $(1/3, 2/3, z)$, $(2/3, 1/3, z+1/2)$ | Same as 2a | Same as 2a |

Here the topological invariant $\widehat{\varphi_3}[T_1, T_2, S_2]$ can be chosen to be the same as that of group No.4 $P2_1$, given by Eq. (148).

## No. 174: $P\bar{6}$

This group is generated by three translations $T_{1,2,3}$ as given in Eqs. (88), a three-fold rotation $C_3$, and a mirror $M$:

$$C_3: (x,y,z) \to (-y, x-y, z), \tag{661a}$$

$$M: (x,y,z) \to (x, y, -z). \tag{661b}$$

The $\mathbb{Z}_2$ cohomology ring is given by

$$\mathbb{Z}_2[A_m, A_z, B_{xy}]/\langle \mathcal{R}_2, \mathcal{R}_4 \rangle \tag{662}$$

where the relations are

$$\mathcal{R}_2: \quad A_z(A_m + A_z), \tag{663a}$$

$$\mathcal{R}_4: \quad B_{xy}^2. \tag{663b}$$

We have the following table regarding IWPs and group cohomology at degree 3.

| Wyckoff position | Little group Intl. | Little group Schönflies | Coordinates | LSM anomaly class | Topo. inv. |
|---|---|---|---|---|---|
| 1a | $\bar{6}$ | $C_{3h}$ | $(0,0,0)$ | $(A_m + A_z)B_{xy}$ | $\varphi_3[T_1, T_2, M]$ |
| 1b | $\bar{6}$ | $C_{3h}$ | $(0,0,1/2)$ | $A_z B_{xy}$ | $\varphi_3[T_1, T_2, T_3 M]$ |
| 1c | $\bar{6}$ | $C_{3h}$ | $(1/3, 2/3, 0)$ | Same as 1a | Same as 1a |
| 1d | $\bar{6}$ | $C_{3h}$ | $(1/3, 2/3, 1/2)$ | Same as 1b | Same as 1b |
| 1e | $\bar{6}$ | $C_{3h}$ | $(2/3, 1/3, 0)$ | Same as 1a | Same as 1a |
| 1f | $\bar{6}$ | $C_{3h}$ | $(2/3, 1/3, 1/2)$ | Same as 1b | Same as 1b |

## No. 175: $P6/m$

This group is generated by three translations $T_{1,2,3}$ as given in Eqs. (88), a three-fold rotation $C_3$, a two-fold rotation $C_2$, and an inversion $I$:

$$C_3: (x,y,z) \to (-y, x-y, z), \tag{664a}$$

$$C_2: (x,y,z) \to (-x, -y, z), \tag{664b}$$

$$I: (x,y,z) \to (-x, -y, -z). \tag{664c}$$

The $\mathbb{Z}_2$ cohomology ring is given by

$$\mathbb{Z}_2[A_i, A_c, A_z, B_{xy}]/\langle \mathcal{R}_2, \mathcal{R}_4 \rangle \tag{665}$$

where the relations are

$$\mathcal{R}_2: \quad A_z(A_i + A_z), \tag{666a}$$

$$\mathcal{R}_4: \quad B_{xy}(A_c^2 + A_i^2 + B_{xy}). \tag{666b}$$

We have the following table regarding IWPs and group cohomology at degree 3.

| Wyckoff position | Little group Intl. | Little group Schönflies | Coordinates | LSM anomaly class | Topo. inv. |
|---|---|---|---|---|---|
| 1a | $6/m$ | $C_{6h}$ | $(0,0,0)$ | $(A_i + A_z)(A_c^2 + A_i^2 + B_{xy})$ | $\varphi_1[I]$ |
| 1b | $6/m$ | $C_{6h}$ | $(0,0,1/2)$ | $A_z(A_c^2 + A_i^2 + B_{xy})$ | $\varphi_1[T_3 I]$ |
| 2c | $\bar{6}$ | $C_{3h}$ | $(1/3, 2/3, 0)$, $(2/3, 1/3, 0)$ | N/A | N/A |
| 2d | $\bar{6}$ | $C_{3h}$ | $(1/3, 2/3, 1/2)$, $(2/3, 1/3, 1/2)$ | N/A | N/A |
| 3f | $2/m$ | $C_{2h}$ | $(1/2, 0, 0)$, $(0, 1/2, 0)$, $(1/2, 1/2, 0)$ | $(A_i + A_z)B_{xy}$ | $\varphi_1[T_1 I]$ |
| 3g | $2/m$ | $C_{2h}$ | $(1/2, 0, 1/2)$, $(0, 1/2, 1/2)$, $(1/2, 1/2, 1/2)$ | $A_z B_{xy}$ | $\varphi_1[T_1 T_3 I]$ |

**No. 176:** $P6_3/m$

This group is generated by three translations $T_{1,2,3}$ as given in Eqs. (88), a three-fold rotation $C_3$, a two-fold screw $S_2$, and an inversion $I$:

$$C_3: (x, y, z) \rightarrow (-y, x - y, z), \tag{667a}$$
$$S_2: (x, y, z) \rightarrow (-x, -y, z + 1/2), \tag{667b}$$
$$I: (x, y, z) \rightarrow (-x, -y, -z). \tag{667c}$$

The $\mathbb{Z}_2$ cohomology ring is given by

$$\mathbb{Z}_2[A_i, A_c, B_{xy}]/\langle \mathcal{R}_2, \mathcal{R}_4 \rangle \tag{668}$$

where the relations are

$$\mathcal{R}_2: \quad A_c(A_c + A_i), \tag{669a}$$
$$\mathcal{R}_4: \quad B_{xy}(A_c A_i + A_i^2 + B_{xy}). \tag{669b}$$

We have the following table regarding IWPs and group cohomology at degree 3.

| Wyckoff position | Little group Intl. | Schönflies | Coordinates | LSM anomaly class | Topo. inv. |
|---|---|---|---|---|---|
| 2a | $\bar{6}$ | $C_{3h}$ | $(0, 0, 1/4), (0, 0, 3/4)$ | $A_c B_{xy}$ | $\varphi_3[T_1, T_2, S_2 I]$ |
| 2b | $\bar{3}$ | $C_{3i}$ | $(0, 0, 0), (0, 0, 1/2)$ | $(A_c + A_i)(A_i^2 + B_{xy})$ | $\varphi_1[I]$ |
| 2c | $\bar{6}$ | $C_{3h}$ | $(1/3, 2/3, 1/4), (2/3, 1/3, 3/4)$ | Same as 2a | Same as 2a |
| 2d | $\bar{6}$ | $C_{3h}$ | $(2/3, 1/3, 1/4), (1/3, 2/3, 3/4)$ | Same as 2a | Same as 2a |
| 6g | $\bar{1}$ | $C_i$ | $(1/2, 0, 0), (0, 1/2, 0),$ $(1/2, 1/2, 0), (1/2, 0, 1/2),$ $(0, 1/2, 1/2), (1/2, 1/2, 1/2)$ | $(A_c + A_i)B_{xy}$ | $\varphi_1[T_1 I]$ |

**No. 177:** $P622$

This group is generated by three translations $T_{1,2,3}$ as given in Eqs. (88), a three-fold rotation $C_3$, a two-fold rotation $C_2'$, and a two-fold rotation $C_2$:

$$C_3: (x, y, z) \rightarrow (-y, x - y, z), \tag{670a}$$
$$C_2': (x, y, z) \rightarrow (y, x, -z), \tag{670b}$$
$$C_2: (x, y, z) \rightarrow (-x, -y, z). \tag{670c}$$

The $\mathbb{Z}_2$ cohomology ring is given by

$$\mathbb{Z}_2[A_c, A_{c'}, A_z, B_{xy}]/\langle \mathcal{R}_2, \mathcal{R}_4 \rangle \tag{671}$$

where the relations are

$$\mathcal{R}_2: \quad A_z(A_{c'} + A_z), \tag{672a}$$
$$\mathcal{R}_4: \quad B_{xy}(A_c^2 + A_c A_{c'} + B_{xy}). \tag{672b}$$

We have the following table regarding IWPs and group cohomology at degree 3.

| Wyckoff position | Little group Intl. | Schönflies | Coordinates | LSM anomaly class | Topo. inv. |
|---|---|---|---|---|---|
| 1a | 622 | $D_6$ | $(0, 0, 0)$ | $(A_{c'} + A_z)(A_c^2 + A_c A_{c'} + B_{xy})$ | $\varphi_2[C_2', C_2]$ |
| 1b | 622 | $D_6$ | $(0, 0, 1/2)$ | $A_z(A_c^2 + A_c A_{c'} + B_{xy})$ | $\varphi_2[T_3 C_2', C_2]$ |
| 2c | 32 | $D_3$ | $(1/3, 2/3, 0), (2/3, 1/3, 0)$ | N/A | N/A |
| 2d | 32 | $D_3$ | $(1/3, 2/3, 1/2), (2/3, 1/3, 1/2)$ | N/A | N/A |
| 3f | 222 | $D_2$ | $(1/2, 0, 0), (0, 1/2, 0),$ $(1/2, 1/2, 0)$ | $(A_{c'} + A_z)B_{xy}$ | $\varphi_2[C_2', T_1 T_2 C_2]$ |
| 3g | 222 | $D_2$ | $(1/2, 0, 1/2), (0, 1/2, 1/2),$ $(1/2, 1/2, 1/2)$ | $A_z B_{xy}$ | $\varphi_2[T_3 C_2', T_1 T_2 C_2]$ |

**No. 178:** $P6_122$

This group is generated by three translations $T_{1,2,3}$ as given in Eqs. (88), a three-fold screw $S_3$, a two-fold rotation $C'_2$, and a two-fold screw $S_2$:

$$S_3\colon (x,y,z) \to (-y, x-y, z+1/3), \tag{673a}$$
$$C'_2\colon (x,y,z) \to (y, x, -z+1/3), \tag{673b}$$
$$S_2\colon (x,y,z) \to (-x, -y, z+1/2). \tag{673c}$$

The $\mathbb{Z}_2$ cohomology ring is given by

$$\mathbb{Z}_2[A_c, A_{c'}, B_{xy}]/\langle \mathscr{R}_2, \mathscr{R}_4 \rangle \tag{674}$$

where the relations are

$$\mathscr{R}_2\colon \quad A_c(A_c + A_{c'}), \tag{675a}$$
$$\mathscr{R}_4\colon \quad B_{xy}^2. \tag{675b}$$

We have the following table regarding IWPs and group cohomology at degree 3.

| Wyckoff position | Little group Intl. | Little group Schönflies | Coordinates | LSM anomaly class | Topo. inv. |
|---|---|---|---|---|---|
| 6a | 2 | $C_2$ | $(x,0,0),\ (0,x,1/3),$ $(-x,-x,2/3),\ (-x,0,1/2),$ $(0,-x,5/6),\ (x,x,1/6)$ | $(A_c + A_{c'})B_{xy}$ | $\varphi_2[T_1T_2, C'_2]$ |
| 6b | 2 | $C_2$ | $(x,2x,1/4),\ (-2x,-x,7/12),$ $(x,-x,11/12),\ (-x,-2x,3/4),$ $(2x,x,1/12),\ (-x,x,5/12)$ | $A_cB_{xy}$ | $\varphi_2[T_1T_2^{-1}, C'_2S_2]$ |

**No. 179:** $P6_522$

This group is generated by three translations $T_{1,2,3}$ as given in Eqs. (88), a three-fold screw $S_3$, a two-fold rotation $C'_2$, and a two-fold screw $S_2$:

$$S_3\colon (x,y,z) \to (-y, x-y, z+2/3), \tag{676a}$$
$$C'_2\colon (x,y,z) \to (y, x, -z+2/3), \tag{676b}$$
$$S_2\colon (x,y,z) \to (-x, -y, z+1/2). \tag{676c}$$

The $\mathbb{Z}_2$ cohomology ring is given by

$$\mathbb{Z}_2[A_c, A_{c'}, B_{xy}]/\langle \mathscr{R}_2, \mathscr{R}_4 \rangle \tag{677}$$

where the relations are

$$\mathscr{R}_2\colon \quad A_c(A_c + A_{c'}), \tag{678a}$$
$$\mathscr{R}_4\colon \quad B_{xy}^2. \tag{678b}$$

We have the following table regarding IWPs and group cohomology at degree 3.

| Wyckoff position | Little group Intl. | Little group Schönflies | Coordinates | LSM anomaly class | Topo. inv. |
|---|---|---|---|---|---|
| 6a | 2 | $C_2$ | $(x,0,0),\ (0,x,2/3),$ $(-x,-x,1/3),\ (-x,0,1/2),$ $(0,-x,1/6),\ (x,x,5/6)$ | $(A_c + A_{c'})B_{xy}$ | $\varphi_2[T_1T_2, C'_2]$ |
| 6b | 2 | $C_2$ | $(x,2x,3/4),\ (-2x,-x,5/12),$ $(x,-x,1/12),\ (-x,-2x,1/4),$ $(2x,x,11/12),\ (-x,x,7/12)$ | $A_cB_{xy}$ | $\varphi_2[T_1T_2^{-1}, C'_2S_2]$ |

**No. 180:** $P6_222$

This group is generated by three translations $T_{1,2,3}$ as given in Eqs. (88), a three-fold screw $S_3$, a two-fold rotation $C_2'$, and a two-fold rotation $C_2$:

$$S_3: (x, y, z) \rightarrow (-y, x - y, z + 2/3), \tag{679a}$$
$$C_2': (x, y, z) \rightarrow (y, x, -z + 2/3), \tag{679b}$$
$$C_2: (x, y, z) \rightarrow (-x, -y, z). \tag{679c}$$

The $\mathbb{Z}_2$ cohomology ring is given by

$$\mathbb{Z}_2[A_c, A_{c'}, A_z, B_{xy}]/\langle \mathscr{R}_2, \mathscr{R}_4 \rangle \tag{680}$$

where the relations are

$$\mathscr{R}_2: \quad A_z(A_{c'} + A_z), \tag{681a}$$
$$\mathscr{R}_4: \quad B_{xy}(A_c^2 + A_c A_{c'} + B_{xy}). \tag{681b}$$

We have the following table regarding IWPs and group cohomology at degree 3.

| Wyckoff position | Little group Intl. | Schönflies | Coordinates | LSM anomaly class | Topo. inv. |
|---|---|---|---|---|---|
| 3a | 222 | $D_2$ | $(0,0,0), (0,0,2/3), (0,0,1/3)$ | $(A_{c'} + A_z)(A_c^2 + A_c A_{c'} + B_{xy})$ | $\varphi_2[C_2', C_2]$ |
| 3b | 222 | $D_2$ | $(0,0,1/2), (0,0,1/6), (0,0,5/6)$ | $A_z(A_c^2 + A_c A_{c'} + B_{xy})$ | $\varphi_2[T_3 C_2', C_2]$ |
| 3c | 222 | $D_2$ | $(1/2,0,0), (0,1/2,2/3), (1/2,1/2,1/3)$ | $(A_{c'} + A_z)B_{xy}$ | $\varphi_2[C_2', T_1 T_2 C_2]$ |
| 3d | 222 | $D_2$ | $(1/2,0,1/2), (0,1/2,1/6), (1/2,1/2,5/6)$ | $A_z B_{xy}$ | $\varphi_2[T_3 C_2', T_1 T_2 C_2]$ |

**No. 181:** $P6_422$

This group is generated by three translations $T_{1,2,3}$ as given in Eqs. (88), a three-fold screw $S_3$, a two-fold rotation $C_2'$, and a two-fold rotation $C_2$:

$$S_3: (x, y, z) \rightarrow (-y, x - y, z + 1/3), \tag{682a}$$
$$C_2': (x, y, z) \rightarrow (y, x, -z + 1/3), \tag{682b}$$
$$C_2: (x, y, z) \rightarrow (-x, -y, z). \tag{682c}$$

The $\mathbb{Z}_2$ cohomology ring is given by

$$\mathbb{Z}_2[A_c, A_{c'}, A_z, B_{xy}]/\langle \mathscr{R}_2, \mathscr{R}_4 \rangle \tag{683}$$

where the relations are

$$\mathscr{R}_2: \quad A_z(A_{c'} + A_z), \tag{684a}$$
$$\mathscr{R}_4: \quad B_{xy}(A_c^2 + A_c A_{c'} + B_{xy}). \tag{684b}$$

We have the following table regarding IWPs and group cohomology at degree 3.

| Wyckoff position | Little group Intl. | Schönflies | Coordinates | LSM anomaly class | Topo. inv. |
|---|---|---|---|---|---|
| 3a | 222 | $D_2$ | $(0,0,0), (0,0,1/3), (0,0,2/3)$ | $A_z(A_c^2 + A_c A_{c'} + B_{xy})$ | $\varphi_2[T_3 C_2', C_2]$ |
| 3b | 222 | $D_2$ | $(0,0,1/2), (0,0,5/6), (0,0,1/6)$ | $(A_{c'} + A_z)(A_c^2 + A_c A_{c'} + B_{xy})$ | $\varphi_2[C_2', C_2]$ |
| 3c | 222 | $D_2$ | $(1/2,0,0), (0,1/2,1/3), (1/2,1/2,2/3)$ | $A_z B_{xy}$ | $\varphi_2[T_3 C_2', T_1 T_2 C_2]$ |
| 3d | 222 | $D_2$ | $(1/2,0,1/2), (0,1/2,5/6), (1/2,1/2,1/6)$ | $(A_{c'} + A_z)B_{xy}$ | $\varphi_2[C_2', T_1 T_2 C_2]$ |

**No. 182:** $P6_322$

This group is generated by three translations $T_{1,2,3}$ as given in Eqs. (88), a three-fold rotation $C_3$, a two-fold rotation $C_2'$, and a two-fold screw $S_2$:

$$C_3 : (x, y, z) \rightarrow (-y, x - y, z), \tag{685a}$$
$$C_2' : (x, y, z) \rightarrow (y, x, -z), \tag{685b}$$
$$S_2 : (x, y, z) \rightarrow (-x, -y, z + 1/2). \tag{685c}$$

The $\mathbb{Z}_2$ cohomology ring is given by

$$\mathbb{Z}_2[A_c, A_{c'}, B_{xy}]/\langle \mathcal{R}_2, \mathcal{R}_4 \rangle \tag{686}$$

where the relations are

$$\mathcal{R}_2 : \quad A_c(A_c + A_{c'}), \tag{687a}$$
$$\mathcal{R}_4 : \quad B_{xy}^2. \tag{687b}$$

We have the following table regarding IWPs and group cohomology at degree 3.

| Wyckoff position | Little group Intl. | Little group Schönflies | Coordinates | LSM anomaly class | Topo. inv. |
|---|---|---|---|---|---|
| 2a | 32 | $D_3$ | $(0,0,0)$, $(0,0,1/2)$ | $(A_c + A_{c'})B_{xy}$ | $\varphi_2[T_1T_2, C_2']$ |
| 2b | 32 | $D_3$ | $(0,0,1/4)$, $(0,0,3/4)$ | $A_c B_{xy}$ | $\varphi_2[T_1T_2^{-1}, C_2'S_2]$ |
| 2c | 32 | $D_3$ | $(1/3,2/3,1/4)$, $(2/3,1/3,3/4)$ | Same as 2b | Same as 2b |
| 2d | 32 | $D_3$ | $(1/3,2/3,3/4)$, $(2/3,1/3,1/4)$ | Same as 2b | Same as 2b |

**No. 183:** $P6mm$

This group is generated by three translations $T_{1,2,3}$ as given in Eqs. (88), a three-fold rotation $C_3$, a mirror $M$, and a two-fold rotation $C_2$:

$$C_3 : (x, y, z) \rightarrow (-y, x - y, z), \tag{688a}$$
$$M : (x, y, z) \rightarrow (-y, -x, z), \tag{688b}$$
$$C_2 : (x, y, z) \rightarrow (-x, -y, z). \tag{688c}$$

The $\mathbb{Z}_2$ cohomology ring is given by

$$\mathbb{Z}_2[A_c, A_m, A_z, B_{xy}]/\langle \mathcal{R}_2, \mathcal{R}_4 \rangle \tag{689}$$

where the relations are

$$\mathcal{R}_2 : \quad A_z^2, \tag{690a}$$
$$\mathcal{R}_4 : \quad B_{xy}(A_c^2 + A_c A_m + B_{xy}). \tag{690b}$$

We have the following table regarding IWPs and group cohomology at degree 3.

| Wyckoff position | Little group Intl. | Little group Schönflies | Coordinates | LSM anomaly class | Topo. inv. |
|---|---|---|---|---|---|
| 1a | $6mm$ | $C_{6v}$ | $(0,0,z)$ | $A_z(A_c^2 + A_c A_m + B_{xy})$ | $\varphi_2[T_3, C_2]$ |
| 2b | $3m$ | $C_{3v}$ | $(1/3,2/3,z)$, $(2/3,1/3,z)$ | N/A | N/A |
| 3c | $2mm$ | $C_{2v}$ | $(1/2,0,z)$, $(0,1/2,z)$, $(1/2,1/2,z)$ | $A_z B_{xy}$ | $\varphi_2[T_3, T_1C_2]$ |

**No. 184:** $P6cc$

This group is generated by three translations $T_{1,2,3}$ as given in Eqs. (88), a three-fold rotation $C_3$, a glide $G$, and a two-fold rotation $C_2$:

$$C_3 : (x, y, z) \rightarrow (-y, x - y, z), \tag{691a}$$
$$G : (x, y, z) \rightarrow (-y, -x, z + 1/2), \tag{691b}$$
$$C_2 : (x, y, z) \rightarrow (-x, -y, z). \tag{691c}$$

The $\mathbb{Z}_2$ cohomology ring is given by

$$\mathbb{Z}_2[A_c, A_m, B_{xy}]/\langle \mathcal{R}_2, \mathcal{R}_4 \rangle \tag{692}$$

where the relations are

$$\mathcal{R}_2: \quad A_m^2, \tag{693a}$$

$$\mathcal{R}_4: \quad B_{xy}(A_c^2 + A_c A_m + B_{xy}). \tag{693b}$$

We have the following table regarding IWPs and group cohomology at degree 3.

| Wyckoff position | Little group Intl. | Little group Schönflies | Coordinates | LSM anomaly class | Topo. inv. |
|---|---|---|---|---|---|
| 2a | 6 | $C_6$ | $(0,0,z)$, $(0,0,z+1/2)$ | $A_m(A_c^2 + B_{xy})$ | $\varphi_2[G, C_2]$ |
| 4b | 3 | $C_3$ | $(1/3, 2/3, z)$, $(2/3, 1/3, z)$, $(1/3, 2/3, z+1/2)$, $(2/3, 1/3, z+1/2)$ | N/A | N/A |
| 6c | 2 | $C_2$ | $(1/2, 0, z)$, $(0, 1/2, z)$, $(1/2, 1/2, z)$, $(0, 1/2, z+1/2)$, $(1/2, 0, z+1/2)$, $(1/2, 1/2, z+1/2)$ | $A_m B_{xy}$ | $\varphi_2[T_1 T_2 G, T_1 T_2 C_2]$ |

**No. 185: $P6_3cm$**

This group is generated by three translations $T_{1,2,3}$ as given in Eqs. (88), a three-fold rotation $C_3$, a glide $G$, and a two-fold screw $S_2$:

$$C_3: (x, y, z) \to (-y, x - y, z), \tag{694a}$$

$$G: (x, y, z) \to (-y, -x, z + 1/2), \tag{694b}$$

$$S_2: (x, y, z) \to (-x, -y, z + 1/2). \tag{694c}$$

The $\mathbb{Z}_2$ cohomology ring is given by

$$\mathbb{Z}_2[A_c, A_m, B_{xy}]/\langle \mathcal{R}_2, \mathcal{R}_4 \rangle \tag{695}$$

where the relations are

$$\mathcal{R}_2: \quad (A_c + A_m)^2, \tag{696a}$$

$$\mathcal{R}_4: \quad B_{xy}(A_c^2 + A_c A_m + B_{xy}). \tag{696b}$$

We have the following table regarding IWPs and group cohomology at degree 3.

| Wyckoff position | Little group Intl. | Little group Schönflies | Coordinates | LSM anomaly class | Topo. inv. |
|---|---|---|---|---|---|
| 2a | 3m | $C_{3v}$ | $(0,0,z)$, $(0,0,z+1/2)$ | $(A_c + A_m)B_{xy}$ | $\widehat{\varphi_3}[T_1, T_2, G]$ |
| 4b | 3 | $C_3$ | $(1/3, 2/3, z)$, $(2/3, 1/3, z+1/2)$, $(1/3, 2/3, z+1/2)$, $(2/3, 1/3, z)$ | N/A | N/A |

Here the topological invariant $\widehat{\varphi_3}[T_1, T_2, G]$ can be chosen to be the same as that of group No.9 $Cc$, given by Eq. (164).

**No. 186: $P6_3mc$**

This group is generated by three translations $T_{1,2,3}$ as given in Eqs. (88), a three-fold rotation $C_3$, a mirror $M$, and a two-fold screw $S_2$:

$$C_3: (x, y, z) \to (-y, x - y, z), \tag{697a}$$

$$M: (x, y, z) \to (-y, -x, z), \tag{697b}$$

$$S_2: (x, y, z) \to (-x, -y, z + 1/2). \tag{697c}$$

The $\mathbb{Z}_2$ cohomology ring is given by

$$\mathbb{Z}_2[A_c, A_m, B_{xy}]/\langle \mathcal{R}_2, \mathcal{R}_4 \rangle \tag{698}$$

where the relations are

$$\mathcal{R}_2: \quad A_c^2, \tag{699a}$$
$$\mathcal{R}_4: \quad B_{xy}(A_c A_m + B_{xy}). \tag{699b}$$

We have the following table regarding IWPs and group cohomology at degree 3.

| Wyckoff position | Little group Intl. | Schönflies | Coordinates | LSM anomaly class | Topo. inv. |
|---|---|---|---|---|---|
| 2a | $3m$ | $C_{3v}$ | $(0,0,z)$, $(0,0,z+1/2)$ | $A_c B_{xy}$ | $\widetilde{\varphi_3}[T_1 T_2^{-1}, MS_2, M]$ |
| 2b | $3m$ | $C_{3v}$ | $(1/3, 2/3, z)$, $(2/3, 1/3, z+1/2)$ | Same as 2a | Same as 2a |

The expression of $\widetilde{\varphi_3}$ is given in Eq. (87).

## No. 187: $P\bar{6}m2$

This group is generated by three translations $T_{1,2,3}$ as given in Eqs. (88), a three-fold rotation $C_3$, a two-fold rotation $C_2'$, and a mirror $M$:

$$C_3: (x,y,z) \to (-y, x-y, z), \tag{700a}$$
$$C_2': (x,y,z) \to (-y, -x, -z), \tag{700b}$$
$$M: (x,y,z) \to (x, y, -z). \tag{700c}$$

The $\mathbb{Z}_2$ cohomology ring is given by

$$\mathbb{Z}_2[A_{c'}, A_m, A_z, B_{xy}]/\langle \mathcal{R}_2, \mathcal{R}_4 \rangle \tag{701}$$

where the relations are

$$\mathcal{R}_2: \quad A_z(A_{c'} + A_m + A_z), \tag{702a}$$
$$\mathcal{R}_4: \quad B_{xy}^2. \tag{702b}$$

We have the following table regarding IWPs and group cohomology at degree 3.

| Wyckoff position | Little group Intl. | Schönflies | Coordinates | LSM anomaly class | Topo. inv. |
|---|---|---|---|---|---|
| 1a | $\bar{6}m2$ | $D_{3h}$ | $(0,0,0)$ | $(A_{c'} + A_m + A_z)B_{xy}$ | $\varphi_2[T_1 T_2^{-1}, C_2']$ |
| 1b | $\bar{6}m2$ | $D_{3h}$ | $(0,0,1/2)$ | $A_z B_{xy}$ | $\varphi_2[T_1 T_2^{-1}, T_3 C_2']$ |
| 1c | $\bar{6}m2$ | $D_{3h}$ | $(1/3, 2/3, 0)$ | Same as 1a | Same as 1a |
| 1d | $\bar{6}m2$ | $D_{3h}$ | $(1/3, 2/3, 1/2)$ | Same as 1b | Same as 1b |
| 1e | $\bar{6}m2$ | $D_{3h}$ | $(2/3, 1/3, 0)$ | Same as 1a | Same as 1a |
| 1f | $\bar{6}m2$ | $D_{3h}$ | $(2/3, 1/3, 1/2)$ | Same as 1b | Same as 1b |

## No. 188: $P\bar{6}c2$

This group is generated by three translations $T_{1,2,3}$ as given in Eqs. (88), a three-fold rotation $C_3$, a two-fold rotation $C_2'$, and a mirror $M$:

$$C_3: (x,y,z) \to (-y, x-y, z), \tag{703a}$$
$$C_2': (x,y,z) \to (-y, -x, -z), \tag{703b}$$
$$M: (x,y,z) \to (x, y, -z+1/2). \tag{703c}$$

The $\mathbb{Z}_2$ cohomology ring is given by

$$\mathbb{Z}_2[A_{c'}, A_m, B_{xy}]/\langle \mathcal{R}_2, \mathcal{R}_4\rangle \tag{704}$$

where the relations are

$$\mathcal{R}_2: \quad A_{c'}A_m, \tag{705a}$$

$$\mathcal{R}_4: \quad B_{xy}^2. \tag{705b}$$

We have the following table regarding IWPs and group cohomology at degree 3.

| Wyckoff position | Little group Intl. | Schönflies | Coordinates | LSM anomaly class | Topo. inv. |
|---|---|---|---|---|---|
| 2a | $32$ | $D_3$ | $(0,0,0)$, $(0,0,1/2)$ | $A_{c'}B_{xy}$ | $\varphi_2[T_1T_2^{-1}, C_2']$ |
| 2b | $\bar{6}$ | $C_{3h}$ | $(0,0,1/4)$, $(0,0,3/4)$ | $A_mB_{xy}$ | $\varphi_3[T_1, T_2, M]$ |
| 2c | $32$ | $D_3$ | $(1/3,2/3,0)$, $(1/3,2/3,1/2)$ | Same as 2a | Same as 2a |
| 2d | $\bar{6}$ | $C_{3h}$ | $(1/3,2/3,1/4)$, $(1/3,2/3,3/4)$ | Same as 2b | Same as 2b |
| 2e | $32$ | $D_3$ | $(2/3,1/3,0)$, $(2/3,1/3,1/2)$ | Same as 2a | Same as 2a |
| 2f | $\bar{6}$ | $C_{3h}$ | $(2/3,1/3,1/4)$, $(2/3,1/3,3/4)$ | Same as 2b | Same as 2b |

### No. 189: $P\bar{6}2m$

This group is generated by three translations $T_{1,2,3}$ as given in Eqs. (88), a three-fold rotation $C_3$, a two-fold rotation $C_2'$, and a mirror $M$:

$$C_3: (x,y,z) \to (-y, x-y, z), \tag{706a}$$

$$C_2': (x,y,z) \to (y, x, -z), \tag{706b}$$

$$M: (x,y,z) \to (x, y, -z). \tag{706c}$$

The $\mathbb{Z}_2$ cohomology ring is given by

$$\mathbb{Z}_2[A_{c'}, A_m, A_z, B_{xy}]/\langle \mathcal{R}_2, \mathcal{R}_4\rangle \tag{707}$$

where the relations are

$$\mathcal{R}_2: \quad A_z(A_{c'} + A_m + A_z), \tag{708a}$$

$$\mathcal{R}_4: \quad B_{xy}^2. \tag{708b}$$

We have the following table regarding IWPs and group cohomology at degree 3.

| Wyckoff position | Little group Intl. | Schönflies | Coordinates | LSM anomaly class | Topo. inv. |
|---|---|---|---|---|---|
| 1a | $\bar{6}2m$ | $D_{3h}$ | $(0,0,0)$ | $(A_{c'} + A_m + A_z)B_{xy}$ | $\varphi_2[T_1T_2, C_2']$ |
| 1b | $\bar{6}2m$ | $D_{3h}$ | $(0,0,1/2)$ | $A_zB_{xy}$ | $\varphi_2[T_1T_2, T_3C_2']$ |
| 2c | $\bar{6}$ | $C_{3h}$ | $(1/3,2/3,0)$, $(2/3,1/3,0)$ | N/A | N/A |
| 2d | $\bar{6}$ | $C_{3h}$ | $(1/3,2/3,1/2)$, $(2/3,1/3,1/2)$ | N/A | N/A |

### No. 190: $P\bar{6}2c$

This group is generated by three translations $T_{1,2,3}$ as given in Eqs. (88), a three-fold rotation $C_3$, a two-fold rotation $C_2'$, and a mirror $M$:

$$C_3: (x,y,z) \to (-y, x-y, z), \tag{709a}$$

$$C_2': (x,y,z) \to (y, x, -z), \tag{709b}$$

$$M: (x,y,z) \to (x, y, -z+1/2). \tag{709c}$$

The $\mathbb{Z}_2$ cohomology ring is given by

$$\mathbb{Z}_2[A_{c'}, A_m, B_{xy}]/\langle \mathscr{R}_2, \mathscr{R}_4\rangle \tag{710}$$

where the relations are

$$\mathscr{R}_2: \quad A_{c'}A_m, \tag{711a}$$

$$\mathscr{R}_4: \quad B_{xy}^2. \tag{711b}$$

We have the following table regarding IWPs and group cohomology at degree 3.

| Wyckoff position | Little group Intl. | Little group Schönflies | Coordinates | LSM anomaly class | Topo. inv. |
|---|---|---|---|---|---|
| 2a | 32 | $D_3$ | $(0,0,0)$, $(0,0,1/2)$ | $A_{c'}B_{xy}$ | $\varphi_2[T_1T_2, C_2']$ |
| 2b | $\bar{6}$ | $C_{3h}$ | $(0,0,1/4)$, $(0,0,3/4)$ | $A_m B_{xy}$ | $\varphi_3[T_1, T_2, M]$ |
| 2c | $\bar{6}$ | $C_{3h}$ | $(1/3,2/3,1/4)$, $(2/3,1/3,3/4)$ | Same as 2a | Same as 2a |
| 2d | $\bar{6}$ | $C_{3h}$ | $(2/3,1/3,1/4)$, $(1/3,2/3,3/4)$ | Same as 2b | Same as 2b |

**No. 191:** $P6/mmm$

This group is generated by three translations $T_{1,2,3}$ as given in Eqs. (88), a three-fold rotation $C_3$, a two-fold rotation $C_2'$, a two-fold rotation $C_2$, and an inversion $I$:

$$C_3: (x,y,z) \to (-y, x-y, z), \tag{712a}$$

$$C_2': (x,y,z) \to (y, x, -z), \tag{712b}$$

$$C_2: (x,y,z) \to (-x, -y, z), \tag{712c}$$

$$I: (x,y,z) \to (-x, -y, -z). \tag{712d}$$

The $\mathbb{Z}_2$ cohomology ring is given by

$$\mathbb{Z}_2[A_i, A_c, A_{c'}, A_z, B_{xy}]/\langle \mathscr{R}_2, \mathscr{R}_4\rangle \tag{713}$$

where the relations are

$$\mathscr{R}_2: \quad A_z(A_{c'} + A_i + A_z), \tag{714a}$$

$$\mathscr{R}_4: \quad B_{xy}(A_c^2 + A_c A_{c'} + A_{c'}A_i + A_i^2 + B_{xy}). \tag{714b}$$

We have the following table regarding IWPs and group cohomology at degree 3.

| Wyckoff position | Little group Intl. | Little group Schönflies | Coordinates | LSM anomaly class | Topo. inv. |
|---|---|---|---|---|---|
| 1a | $6/mmm$ | $D_{6h}$ | $(0,0,0)$ | $(A_{c'} + A_i + A_z)(A_c^2 + A_c A_{c'} + A_{c'}A_i + A_i^2 + B_{xy})$ | $\varphi_2[C_2', C_2]$ |
| 1b | $6/mmm$ | $D_{6h}$ | $(0,0,1/2)$ | $A_z(A_c^2 + A_c A_{c'} + A_{c'}A_i + A_i^2 + B_{xy})$ | $\varphi_2[T_3C_2', C_2]$ |
| 2c | $\bar{6}m2$ | $D_{3h}$ | $(1/3,2/3,0)$, $(2/3,1/3,0)$ | N/A | N/A |
| 2d | $\bar{6}m2$ | $D_{3h}$ | $(1/3,2/3,1/2)$, $(2/3,1/3,1/2)$ | N/A | N/A |
| 3f | $mmm$ | $D_{2h}$ | $(1/2,0,0)$, $(0,1/2,0)$, $(1/2,1/2,0)$ | $(A_{c'} + A_i + A_z)B_{xy}$ | $\varphi_2[C_2', T_1T_2C_2]$ |
| 3g | $mmm$ | $D_{2h}$ | $(1/2,0,1/2)$, $(0,1/2,1/2)$, $(1/2,1/2,1/2)$ | $A_z B_{xy}$ | $\varphi_2[T_3C_2', T_1T_2C_2]$ |

**No. 192:** $P6/mcc$

This group is generated by three translations $T_{1,2,3}$ as given in Eqs. (88), a three-fold rotation $C_3$, a two-fold rotation $C_2'$, a two-fold rotation $C_2$, and an inversion $I$:

$$C_3: (x,y,z) \to (-y, x-y, z), \tag{715a}$$

$$C_2': (x,y,z) \to (y, x, -z+1/2), \tag{715b}$$

$$C_2: (x,y,z) \to (-x, -y, z), \tag{715c}$$

$$I: (x,y,z) \to (-x, -y, -z). \tag{715d}$$

The $\mathbb{Z}_2$ cohomology ring is given by

$$\mathbb{Z}_2[A_i, A_c, A_{c'}, B_{xy}]/\langle \mathscr{R}_2, \mathscr{R}_4 \rangle \tag{716}$$

where the relations are

$$\mathscr{R}_2: \quad A_{c'}A_i, \tag{717a}$$
$$\mathscr{R}_4: \quad B_{xy}(A_c^2 + A_cA_{c'} + A_i^2 + B_{xy}). \tag{717b}$$

We have the following table regarding IWPs and group cohomology at degree 3.

| Wyckoff position | Little group Intl. | Schönflies | Coordinates | LSM anomaly class | Topo. inv. |
|---|---|---|---|---|---|
| 2a | 622 | $D_6$ | $(0,0,1/4)$, $(0,0,3/4)$ | $A_{c'}(A_c^2 + A_cA_{c'} + B_{xy})$ | $\varphi_2[C_2', C_2]$ |
| 2b | $6/m$ | $C_{6h}$ | $(0,0,0)$, $(0,0,1/2)$ | $A_i(A_c^2 + A_i^2 + B_{xy})$ | $\varphi_1[I]$ |
| 4c | 32 | $D_3$ | $(1/3,2/3,1/4)$, $(2/3,1/3,1/4)$, $(2/3,1/3,3/4)$, $(1/3,2/3,3/4)$ | N/A | N/A |
| 4d | $\bar{6}$ | $C_{3h}$ | $(1/3,2/3,0)$, $(2/3,1/3,0)$, $(2/3,1/3,1/2)$, $(1/3,2/3,1/2)$ | N/A | N/A |
| 6f | 222 | $D_2$ | $(1/2,0,1/4)$, $(0,1/2,1/4)$, $(1/2,1/2,1/4)$, $(1/2,0,3/4)$, $(0,1/2,3/4)$, $(1/2,1/2,3/4)$ | $A_{c'}B_{xy}$ | $\varphi_2[C_2', T_1T_2C_2]$ |
| 6g | $2/m$ | $C_{2h}$ | $(1/2,0,0)$, $(0,1/2,0)$, $(1/2,1/2,0)$, $(0,1/2,1/2)$, $(1/2,0,1/2)$, $(1/2,1/2,1/2)$ | $A_iB_{xy}$ | $\varphi_1[T_1I]$ |

**No. 193:** $P6_3/mcm$

This group is generated by three translations $T_{1,2,3}$ as given in Eqs. (88), a three-fold rotation $C_3$, a two-fold rotation $C_2'$, a two-fold screw $S_2$, and an inversion $I$:

$$C_3: (x,y,z) \rightarrow (-y, x-y, z), \tag{718a}$$
$$C_2': (x,y,z) \rightarrow (y, x, -z+1/2), \tag{718b}$$
$$S_2: (x,y,z) \rightarrow (-x, -y, z+1/2), \tag{718c}$$
$$I: (x,y,z) \rightarrow (-x, -y, -z). \tag{718d}$$

The $\mathbb{Z}_2$ cohomology ring is given by

$$\mathbb{Z}_2[A_i, A_c, A_{c'}, B_{xy}]/\langle \mathscr{R}_2, \mathscr{R}_4 \rangle \tag{719}$$

where the relations are

$$\mathscr{R}_2: \quad (A_c + A_{c'})(A_c + A_i), \tag{720a}$$
$$\mathscr{R}_4: \quad B_{xy}(A_cA_i + A_i^2 + B_{xy}). \tag{720b}$$

We have the following table regarding IWPs and group cohomology at degree 3.

| Wyckoff position | Little group Intl. | Schönflies | Coordinates | LSM anomaly class | Topo. inv. |
|---|---|---|---|---|---|
| 2a | $\bar{6}2m$ | $D_{3h}$ | $(0,0,1/4)$, $(0,0,3/4)$ | $(A_c + A_{c'})B_{xy}$ | $\varphi_2[T_1T_2, C_2']$ |
| 2b | $\bar{3}m$ | $D_{3d}$ | $(0,0,0)$, $(0,0,1/2)$ | $(A_c + A_i)(A_cA_i + A_i^2 + B_{xy})$ | $\varphi_1[I]$ |
| 4c | $\bar{6}$ | $C_{3h}$ | $(1/3,2/3,1/4)$, $(2/3,1/3,3/4)$, $(2/3,1/3,1/4)$, $(1/3,2/3,3/4)$ | N/A | N/A |
| 4d | 32 | $D_3$ | $(1/3,2/3,0)$, $(2/3,1/3,1/2)$, $(2/3,1/3,0)$, $(1/3,2/3,1/2)$ | N/A | N/A |
| 6f | $2/m$ | $C_{2h}$ | $(1/2,0,0)$, $(0,1/2,0)$, $(1/2,1/2,0)$, $(1/2,0,1/2)$, $(0,1/2,1/2)$, $(1/2,1/2,1/2)$ | $(A_c + A_i)B_{xy}$ | $\varphi_1[T_1I]$ |

**No. 194:** $P6_3/mmc$

This group is generated by three translations $T_{1,2,3}$ as given in Eqs. (88), a three-fold rotation $C_3$, a two-fold rotation $C_2'$, a two-fold screw $S_2$, and an inversion $I$:

$$C_3: (x, y, z) \rightarrow (-y, x - y, z), \tag{721a}$$
$$C_2': (x, y, z) \rightarrow (y, x, -z), \tag{721b}$$
$$S_2: (x, y, z) \rightarrow (-x, -y, z + 1/2), \tag{721c}$$
$$I: (x, y, z) \rightarrow (-x, -y, -z). \tag{721d}$$

The $\mathbb{Z}_2$ cohomology ring is given by

$$\mathbb{Z}_2[A_i, A_c, A_{c'}, B_{xy}]/\langle \mathcal{R}_2, \mathcal{R}_4 \rangle \tag{722}$$

where the relations are

$$\mathcal{R}_2: \quad A_c(A_c + A_{c'} + A_i), \tag{723a}$$
$$\mathcal{R}_4: \quad B_{xy}(A_c A_i + A_{c'} A_i + A_i^2 + B_{xy}). \tag{723b}$$

We have the following table regarding IWPs and group cohomology at degree 3.

| Wyckoff position | Little group Intl. | Little group Schönflies | Coordinates | LSM anomaly class | Topo. inv. |
|---|---|---|---|---|---|
| 2a | $\bar{3}m$ | $D_{3d}$ | $(0,0,0)$, $(0,0,1/2)$ | $(A_c + A_{c'} + A_i)(A_c A_i + A_{c'} A_i + A_i^2 + B_{xy})$ | $\varphi_1[I]$ |
| 2b | $\bar{6}m2$ | $D_{3h}$ | $(0,0,1/4)$, $(0,0,3/4)$ | $A_c B_{xy}$ | $\varphi_3[T_1, T_2, S_2 I]$ |
| 2c | $\bar{6}m2$ | $D_{3h}$ | $(1/3, 2/3, 1/4)$, $(2/3, 1/3, 3/4)$ | Same as 2b | Same as 2b |
| 2d | $\bar{6}m2$ | $D_{3h}$ | $(1/3, 2/3, 3/4)$, $(2/3, 1/3, 1/4)$ | Same as 2b | Same as 2b |
| 6g | $2m$ | $C_{2h}$ | $(1/2, 0, 0)$, $(0, 1/2, 0)$, $(1/2, 1/2, 0)$, $(1/2, 0, 1/2)$, $(0, 1/2, 1/2)$, $(1/2, 1/2, 1/2)$ | $(A_c + A_{c'} + A_i)B_{xy}$ | $\varphi_1[T_1 I]$ |

**No. 195:** $P23$

This group is generated by three translations $T_{1,2,3}$ as given in Eqs. (88), a two-fold rotation $C_2$, a two-fold rotation $C_2'$, and a three-fold rotation $C_3$:

$$C_2: (x, y, z) \rightarrow (-x, -y, z), \tag{724a}$$
$$C_2': (x, y, z) \rightarrow (-x, y, -z), \tag{724b}$$
$$C_3: (x, y, z) \rightarrow (z, x, y). \tag{724c}$$

The $\mathbb{Z}_2$ cohomology ring is given by

$$\mathbb{Z}_2[A_{x+y+z}, B_\alpha, B_\beta, B_{xy+xz+yz}, C_{\alpha 1}, C_{\alpha 2}, C_\beta, C_{xyz}]/\langle \mathcal{R}_4, \mathcal{R}_5, \mathcal{R}_6 \rangle \tag{725}$$

where the relations are

$\mathcal{R}_4$: $A_{x+y+z}(A^3_{x+y+z} + A_{x+y+z}B_\alpha + C_{\alpha 1} + C_{\alpha 2})$, $A_{x+y+z}C_{xyz}$,

$\qquad A^4_{x+y+z} + A^2_{x+y+z}B_\beta + B_\alpha B_\beta + B_\alpha B_{xy+xz+yz} + A_{x+y+z}C_{\alpha 1}$, $A^2_{x+y+z}B_\alpha + B_\alpha B_\beta + B^2_\beta + A_{x+y+z}C_{\alpha 1}$,

$\qquad A^4_{x+y+z} + A^2_{x+y+z}B_\alpha + B_\alpha B_\beta + B_\beta B_{xy+xz+yz} + A_{x+y+z}C_{\alpha 1} + A_{x+y+z}C_\beta$,

$\qquad A^4_{x+y+z} + A^2_{x+y+z}B_\beta + B_\alpha B_\beta + A^2_{x+y+z}B_{xy+xz+yz} + B^2_{xy+xz+yz} + A_{x+y+z}C_{\alpha 1}$, $\qquad\qquad\qquad\qquad$ (726a)

$\mathcal{R}_5$: $A^3_{x+y+z}B_\alpha + A_{x+y+z}B^2_\alpha + A^3_{x+y+z}B_{xy+xz+yz} + A^2_{x+y+z}C_{\alpha 1} + B_\beta C_{\alpha 1} + B_{xy+xz+yz}C_{\alpha 1} + A^2_{x+y+z}C_\beta$,

$\qquad A^5_{x+y+z} + A^3_{x+y+z}B_\alpha + A_{x+y+z}B^2_\alpha + B_\beta C_{\alpha 2} + B_{xy+xz+yz}C_{\alpha 2} + A^2_{x+y+z}C_\beta$, $A^2_{x+y+z}C_{\alpha 1} + B_\beta C_{\alpha 2} + B_\alpha C_\beta$,

$\qquad A^5_{x+y+z} + A_{x+y+z}B^2_\alpha + A^3_{x+y+z}B_{xy+xz+yz} + A^2_{x+y+z}C_{\alpha 1} + B_\beta C_{\alpha 2} + A^2_{x+y+z}C_\beta + B_\beta C_\beta$,

$\qquad A_{x+y+z}B^2_\alpha + A_{x+y+z}B_\alpha B_\beta + A^2_{x+y+z}C_{\alpha 1} + B_\beta C_{\alpha 2} + B_{xy+xz+yz}C_\beta$,

$\qquad (B_\alpha + B_{xy+xz+yz})C_{xyz}$, $B_\beta(A^3_{x+y+z} + A_{x+y+z}B_\alpha + C_{\alpha 1} + C_{\alpha 2} + C_{xyz})$,

$\qquad A^3_{x+y+z}B_\beta + A_{x+y+z}B_\alpha B_\beta + B_\beta C_{\alpha 1} + B_\beta C_{\alpha 2} + B_{xy+xz+yz}C_{xyz}$, $\qquad\qquad\qquad\qquad\qquad\qquad$ (726b)

$\mathcal{R}_6$: $B^3_\alpha + C^2_{\alpha 1} + C_{\alpha 1}C_{\alpha 2} + C^2_{\alpha 2}$, $C_{\alpha 1}C_\beta + C_{\alpha 1}C_{xyz} + C_\beta C_{xyz} + C^2_{xyz}$,

$\qquad A^4_{x+y+z}B_\alpha + A^2_{x+y+z}B^2_\alpha + B^2_\alpha B_\beta + A^4_{x+y+z}B_{xy+xz+yz} + A_{x+y+z}B_\beta C_{\alpha 1} + A^3_{x+y+z}C_\beta + C_{\alpha 2}C_\beta + C_{\alpha 1}C_{xyz}$,

$\qquad A^4_{x+y+z}B_\alpha + A^2_{x+y+z}B^2_\alpha + A^4_{x+y+z}B_\beta + A^2_{x+y+z}B_\alpha B_\beta + B^2_\alpha B_\beta + A^4_{x+y+z}B_{xy+xz+yz} + A^3_{x+y+z}C_{\alpha 1} + C_{\alpha 1}C_\beta$

$\qquad + C^2_{xyz}$. $\qquad\qquad\qquad\qquad\qquad\qquad\qquad\qquad\qquad\qquad\qquad\qquad\qquad\qquad\qquad\qquad\qquad\qquad\qquad\qquad\qquad\qquad$ (726c)

We have the following table regarding IWPs and group cohomology at degree 3.

| Wyckoff position | Little group Intl. | Little group Schönflies | Coordinates | LSM anomaly class | Topo. inv. |
|---|---|---|---|---|---|
| 1a | 23 | $T$ | $(0,0,0)$ | $A^3_{x+y+z} + A_{x+y+z}B_\alpha + C_{\alpha 1} + C_{\alpha 2} + C_{xyz}$ | $\varphi_2[C_2, C'_2]$ |
| 1b | 23 | $T$ | $(1/2,1/2,1/2)$ | $C_{xyz}$ | $\varphi_2[T_1T_2C_2, T_1T_3C'_2]$ |
| 3c | 222 | $D_2$ | $(0,1/2,1/2), (1/2,0,1/2),$ $(1/2,1/2,0)$ | $A_{x+y+z}B_{xy+xz+yz}$ | $\varphi_2[T_1T_2C_2, T_1C'_2]$ |
| 3d | 222 | $D_2$ | $(1/2,0,0), (0,1/2,0),$ $(0,0,1/2)$ | $A_{x+y+z}(A^2_{x+y+z} + B_\alpha + B_{xy+xz+yz})$ | $\varphi_2[C_2, T_3C'_2]$ |

**No. 196:** $F23$

This group is generated by three translations $T_{1,2,3}$ as given in Eqs. (92), a two-fold rotation $C_2$, a two-fold rotation $C'_2$, and a three-fold rotation $C_3$:

$$C_2\colon (x,y,z) \to (-x,-y,z), \qquad\qquad\qquad (727a)$$
$$C'_2\colon (x,y,z) \to (-x,y,-z), \qquad\qquad\qquad (727b)$$
$$C_3\colon (x,y,z) \to (z,x,y). \qquad\qquad\qquad (727c)$$

The $\mathbb{Z}_2$ cohomology ring is given by

$$\mathbb{Z}_2[B_\alpha, B_\beta, B_{xy+xz+yz}, C_{\alpha 1}, C_{\alpha 2}, C_{\beta 1}, C_{\beta 2}, C_\gamma, C_{xyz}]/\langle \mathcal{R}_4, \mathcal{R}_5, \mathcal{R}_6 \rangle \qquad\qquad (728)$$

where the relations are

$\mathcal{R}_4$: $B_\alpha B_{xy+xz+yz}$, $B_\beta(B_\alpha + B_\beta)$, $B_\beta B_{xy+xz+yz}$, $B^2_{xy+xz+yz}$, $\qquad\qquad\qquad\qquad\qquad\qquad\qquad\qquad\qquad\qquad$ (729a)

$\mathcal{R}_5$: $B_{xy+xz+yz}C_{\alpha 1}$, $B_{xy+xz+yz}C_{\alpha 2}$, $B_\beta C_{\alpha 1} + B_\beta C_{\alpha 2} + B_\alpha C_{\beta 1}$, $B_\beta(C_{\alpha 1} + C_{\alpha 2} + C_{\beta 1})$, $B_{xy+xz+yz}C_{\beta 1}$,

$\qquad B_\beta C_{\alpha 2} + B_\alpha C_{\beta 2}$, $B_\beta(C_{\alpha 2} + C_{\beta 2})$, $B_{xy+xz+yz}C_{\beta 2}$, $B_{xy+xz+yz}C_\gamma$, $B_\beta C_\gamma + B_\alpha C_{xyz}$,

$\qquad B_\beta(C_\gamma + C_{xyz})$, $B_{xy+xz+yz}C_{xyz}$, $\qquad\qquad\qquad\qquad\qquad\qquad\qquad\qquad\qquad\qquad\qquad\qquad\qquad\qquad\qquad\qquad$ (729b)

$\mathcal{R}_6$: $B^3_\alpha + C^2_{\alpha 1} + C_{\alpha 1}C_{\alpha 2} + C^2_{\alpha 2}$, $B^2_\alpha B_\beta + C_{\alpha 1}C_{\beta 1} + C_{\alpha 2}C_{\beta 1} + C_{\alpha 1}C_{\beta 2}$, $B^2_\alpha B_\beta + C_{\alpha 1}C_{\beta 1} + C_{\alpha 2}C_{\beta 2}$,

$\qquad B^2_\alpha B_\beta + C^2_{\beta 1} + C_{\alpha 1}C_{\beta 2}$, $B^2_\alpha B_\beta + C_{\alpha 1}C_{\beta 1} + C_{\alpha 1}C_{\beta 2} + C_{\beta 1}C_{\beta 2}$, $C_{\beta 1}C_\gamma + C_{\alpha 1}C_{xyz} + C_{\alpha 2}C_{xyz}$,

$\qquad (C_{\alpha 1} + C_{\alpha 2} + C_{\beta 1})C_{xyz}$, $B^2_\alpha B_\beta + C_{\alpha 1}C_{\beta 1} + C^2_{\beta 2}$, $C_{\beta 2}C_\gamma + C_{\alpha 2}C_{xyz}$, $(C_{\alpha 2} + C_{\beta 2})C_{xyz}$, $C_\gamma(C_{\alpha 1} + C_{\alpha 2} + C_\gamma)$,

$\qquad (C_{\alpha 1} + C_{\alpha 2} + C_\gamma)C_{xyz}$, $C_{xyz}(C_{\alpha 1} + C_{\alpha 2} + C_{xyz})$. $\qquad\qquad\qquad\qquad\qquad\qquad\qquad\qquad\qquad\qquad\qquad\qquad$ (729c)

We have the following table regarding IWPs and group cohomology at degree 3.

| Wyckoff position | Little group Intl. | Schönflies | Coordinates $(0,0,0)+$ $(0,1/2,1/2)+$ $(1/2,0,1/2)+$ $(1/2,1/2,0)+$ | LSM anomaly class | Topo. inv. |
|---|---|---|---|---|---|
| 4a | 23 | $T$ | $(0,0,0)$ | $C_{\alpha 1} + C_{\alpha 2} + C_{\beta 1} + C_\gamma + C_{xyz}$ | $\varphi_2[C_2, C_2']$ |
| 4b | 23 | $T$ | $(1/2,1/2,1/2)$ | $C_\gamma + C_{xyz}$ | $\varphi_2[C_2, T_1 T_2 T_3^{-1} C_2']$ |
| 4c | 23 | $T$ | $(1/4,1/4,1/4)$ | $C_{xyz}$ | $\varphi_2[T_3 C_2, T_2 C_2']$ |
| 4d | 23 | $T$ | $(3/4,3/4,3/4)$ | $C_{\beta 1} + C_{xyz}$ | $\varphi_2[T_3^{-1} C_2, T_2^{-1} C_2']$ |

## No. 197: $I23$

This group is generated by three translations $T_{1,2,3}$ as given in Eqs. (91), a two-fold rotation $C_2$, a two-fold rotation $C_2'$, and a three-fold rotation $C_3$:

$$C_2 \colon (x,y,z) \to (-x,-y,z), \tag{730a}$$
$$C_2' \colon (x,y,z) \to (-x,y,-z), \tag{730b}$$
$$C_3 \colon (x,y,z) \to (z,x,y). \tag{730c}$$

The $\mathbb{Z}_2$ cohomology ring is given by

$$\mathbb{Z}_2[A_{x+y+z}, B_\alpha, B_\beta, C_{\alpha 1}, C_{\alpha 2}, C_\gamma, C_{xyz}, D_\gamma]/\langle \mathcal{R}_2, \mathcal{R}_3, \mathcal{R}_4, \mathcal{R}_5, \mathcal{R}_6 \rangle \tag{731}$$

where the relations are

$$\mathcal{R}_2 \colon \quad A_{x+y+z}^2, \tag{732a}$$
$$\mathcal{R}_3 \colon \quad A_{x+y+z} B_\alpha, \quad A_{x+y+z} B_\beta, \tag{732b}$$
$$\mathcal{R}_4 \colon \quad A_{x+y+z} C_{\alpha 1}, \quad A_{x+y+z} C_{\alpha 2}, \quad A_{x+y+z} C_\gamma, \quad A_{x+y+z} C_{xyz}, \tag{732c}$$
$$\mathcal{R}_5 \colon \quad A_{x+y+z} D_\gamma, \quad B_\alpha C_\gamma + B_\beta C_\gamma + B_\alpha C_{xyz}, \quad B_\beta(C_{\alpha 1} + C_{\alpha 2} + C_{xyz}), \tag{732d}$$

$$\mathcal{R}_6 \colon \quad B_\alpha^2 B_\beta + B_\alpha B_\beta^2 + C_{\alpha 1} C_\gamma + B_\alpha D_\gamma, \quad B_\alpha^2 B_\beta + B_\beta^3 + C_{\alpha 1} C_{xyz} + B_\alpha D_\gamma + B_\beta D_\gamma, \quad B_\alpha^3 + C_{\alpha 1}^2 + C_{\alpha 1} C_{\alpha 2} + C_{\alpha 2}^2,$$
$$C_{\alpha 2} C_\gamma + B_\alpha D_\gamma, \quad C_{\alpha 2} C_{xyz} + B_\alpha D_\gamma + B_\beta D_\gamma, \quad B_\alpha^2 B_\beta + C_\gamma^2, \quad B_\alpha^2 B_\beta + B_\alpha B_\beta^2 + C_\gamma C_{xyz}, \quad B_\alpha^2 B_\beta + B_\beta^3 + C_{xyz}^2, \tag{732e}$$

$$\mathcal{R}_7 \colon \quad B_\alpha B_\beta C_{\alpha 1} + B_\beta^2 C_{\alpha 1} + B_\alpha B_\beta C_{\alpha 2} + B_\beta^2 C_{\alpha 2} + B_\alpha^2 C_\gamma + C_{\alpha 1} D_\gamma, \quad B_\alpha B_\beta C_{\alpha 1} + B_\beta^2 C_{\alpha 1} + B_\beta^2 C_\gamma + C_{\alpha 2} D_\gamma,$$
$$B_\alpha B_\beta C_{\alpha 2} + C_\gamma D_\gamma, \quad B_\alpha B_\beta C_{\alpha 2} + B_\beta^2 C_{\alpha 2} + C_{xyz} D_\gamma, \tag{732f}$$
$$\mathcal{R}_8 \colon \quad B_\alpha^3 B_\beta + B_\alpha^2 B_\beta^2 + B_\beta^4 + B_\alpha B_\beta D_\gamma + B_\beta^2 D_\gamma + D_\gamma^2. \tag{732g}$$

We have the following table regarding IWPs and group cohomology at degree 3.

| Wyckoff position | Little group Intl. | Schönflies | Coordinates $(0,0,0)+$ $(1/2,1/2,1/2)+$ | LSM anomaly class | Topo. inv. |
|---|---|---|---|---|---|
| 2a | 23 | $T$ | $(0,0,0)$ | $C_{\alpha 1} + C_{\alpha 2} + C_{xyz}$ | $\varphi_2[C_2, C_2']$ |
| 6b | 222 | $D_2$ | $(0,1/2,1/2), (1/2,0,1/2),$ $(1/2,1/2,0)$ | $C_{xyz}$ | $\varphi_2[C_2, T_1 T_2 C_2']$ |

## No. 198: $P2_13$

This group is generated by three translations $T_{1,2,3}$ as given in Eqs. (88), a two-fold screw $S_2$, a two-fold screw $S_2'$, and a three-fold rotation $C_3$:

$$S_2 \colon (x,y,z) \to (-x+1/2, -y, z+1/2), \tag{733a}$$
$$S_2' \colon (x,y,z) \to (-x, y+1/2, -z+1/2), \tag{733b}$$
$$C_3 \colon (x,y,z) \to (z,x,y). \tag{733c}$$

The $\mathbb{Z}_2$ cohomology ring is given by

$$\mathbb{Z}_2[C_\beta]/\langle \mathcal{R}_6 \rangle \tag{734}$$

where the relations are

$$\mathcal{R}_6: \quad C_\beta^2. \tag{735a}$$

We have the following table regarding IWPs and group cohomology at degree 3.

| Wyckoff position | Little group Intl. | Little group Schönflies | Coordinates | LSM anomaly class | Topo. inv. |
|---|---|---|---|---|---|
| 4a | 3 | $C_3$ | $(x,x,x)$, $(-x+1/2,-x,x+1/2)$, $(-x,x+1/2,-x+1/2)$, $(x+1/2,-x+1/2,-x)$ | $C_\beta$ | $\widehat{\varphi_4}[T_2,T_3,S_2,S_2']$ |

Here the topological invariant $\widehat{\varphi_4}[T_2,T_3,S_2,S_2']$ can be chosen to be the same as that of group No.19 $P2_12_12_1$, given by Eq. (195).

## No. 199: $I2_13$

This group is generated by three translations $T_{1,2,3}$ as given in Eqs. (91), a two-fold rotation $C_2$, a two-fold rotation $C_2'$, and a three-fold rotation $C_3$:

$$C_2\colon (x,y,z) \to (-x,-y+1/2,z), \tag{736a}$$
$$C_2'\colon (x,y,z) \to (-x+1/2,y,-z), \tag{736b}$$
$$C_3\colon (x,y,z) \to (z,x,y). \tag{736c}$$

The $\mathbb{Z}_2$ cohomology ring is given by

$$\mathbb{Z}_2[A_{x+y+z},C_\gamma]/\langle\mathcal{R}_6\rangle \tag{737}$$

where the relations are

$$\mathcal{R}_6: \quad C_\gamma^2. \tag{738a}$$

We have the following table regarding IWPs and group cohomology at degree 3.

| Wyckoff position | Little group Intl. | Little group Schönflies | Coordinates $(0,0,0)+ \ (1/2,1/2,1/2)+$ | LSM anomaly class | Topo. inv. |
|---|---|---|---|---|---|
| 8a | 3 | $C_3$ | $(x,x,x)$, $(-x+1/2,-x,x+1/2)$, $(-x,x+1/2,-x+1/2)$, $(x+1/2,-x+1/2,-x)$ | N/A | N/A |
| 12b | 2 | $C_2$ | $(x,0,1/4)$, $(-x+1/2,0,3/4)$, $(1/4,x,0)$, $(3/4,-x+1/2,0)$, $(0,1/4,x)$, $(0,3/4,-x+1/2)$ | $C_\gamma$ | $\varphi_2[T_1T_2,C_2]$ |

## No. 200: $Pm\overline{3}$

This group is generated by three translations $T_{1,2,3}$ as given in Eqs. (88), a two-fold rotation $C_2$, a two-fold rotation $C_2'$, a three-fold rotation $C_3$, and an inversion $I$:

$$C_2\colon (x,y,z) \to (-x,-y,z), \tag{739a}$$
$$C_2'\colon (x,y,z) \to (-x,y,-z), \tag{739b}$$
$$C_3\colon (x,y,z) \to (z,x,y), \tag{739c}$$
$$I\colon (x,y,z) \to (-x,-y,-z). \tag{739d}$$

The $\mathbb{Z}_2$ cohomology ring is given by

$$\mathbb{Z}_2[A_i,A_{x+y+z},B_\alpha,B_\beta,B_{xy+xz+yz},C_{\alpha 1},C_{\alpha 2},C_\beta,C_{xyz}]/\langle\mathcal{R}_4,\mathcal{R}_5,\mathcal{R}_6\rangle \tag{740}$$

where the relations are

$\mathcal{R}_4:$ $A_i^3 A_{x+y+z} + A_i^2 A_{x+y+z}^2 + A_i A_{x+y+z}^3 + A_{x+y+z}^4 + A_i A_{x+y+z} B_\alpha + A_{x+y+z}^2 B_\alpha$

$\quad + A_i A_{x+y+z} B_{xy+xz+yz} + A_{x+y+z} C_{\alpha 1} + A_{x+y+z} C_{\alpha 2} + A_i C_{xyz},$

$\quad A_{x+y+z}(A_i^3 + A_i^2 A_{x+y+z} + A_i A_{x+y+z}^2 + A_{x+y+z}^3 + A_i B_\alpha + A_{x+y+z} B_\alpha + A_i B_{xy+xz+yz} + C_{\alpha 1} + C_{\alpha 2} + C_{xyz}),$

$\quad A_i^2 A_{x+y+z}^2 + A_i A_{x+y+z}^3 + A_i A_{x+y+z} B_\alpha + A_{x+y+z}^2 B_\alpha + A_i A_{x+y+z} B_\beta + A_{x+y+z}^2 B_\beta$

$\quad + B_\alpha B_\beta + B_\alpha B_{xy+xz+yz} + A_{x+y+z} C_{\alpha 2} + A_i C_\beta,$

$\quad A_{x+y+z}^4 + A_i A_{x+y+z} B_\alpha + B_\alpha B_\beta + B_\beta^2 + A_{x+y+z} C_{\alpha 2} + A_i C_\beta,$

$\quad A_i^3 A_{x+y+z} + A_i^2 A_{x+y+z}^2 + A_i^2 B_\beta + A_i A_{x+y+z} B_\beta + B_\alpha B_\beta + A_i A_{x+y+z} B_{xy+xz+yz}$

$\quad + B_\beta B_{xy+xz+yz} + A_{x+y+z} C_{\alpha 2} + A_{x+y+z} C_\beta,$

$\quad A_i^2 A_{x+y+z}^2 + A_i A_{x+y+z}^3 + A_i A_{x+y+z} B_\alpha + A_{x+y+z}^2 B_\alpha + A_i A_{x+y+z} B_\beta + A_{x+y+z}^2 B_\beta + B_\alpha B_\beta + A_i^2 B_{xy+xz+yz}$

$\quad + A_i A_{x+y+z} B_{xy+xz+yz} + A_{x+y+z}^2 B_{xy+xz+yz} + B_{xy+xz+yz}^2 + A_{x+y+z} C_{\alpha 2} + A_i C_\beta,$ $\qquad$ (741a)

$\mathcal{R}_5:$ $A_i^4 A_{x+y+z} + A_i^3 A_{x+y+z}^2 + A_i^2 A_{x+y+z}^3 + A_i A_{x+y+z}^4 + A_i^2 A_{x+y+z} B_\alpha + A_i A_{x+y+z}^2 B_\alpha$

$\quad + A_{x+y+z}^3 B_\alpha + A_{x+y+z} B_\alpha^2 + A_i B_\alpha B_\beta + A_i^2 A_{x+y+z} B_{xy+xz+yz} + A_{x+y+z}^3 B_{xy+xz+yz}$

$\quad + A_i A_{x+y+z} C_{\alpha 1} + A_{x+y+z}^2 C_{\alpha 1} + B_\beta C_{\alpha 1} + B_{xy+xz+yz} C_{\alpha 1} + A_i A_{x+y+z} C_\beta + A_{x+y+z}^2 C_\beta,$

$\quad A_i^4 A_{x+y+z} + A_{x+y+z}^5 + A_{x+y+z}^3 B_\alpha + A_{x+y+z} B_\alpha^2 + A_i B_\alpha B_\beta + A_i^2 A_{x+y+z} B_{xy+xz+yz} + A_i A_{x+y+z}^2 B_{xy+xz+yz}$

$\quad + B_\beta C_{\alpha 2} + B_{xy+xz+yz} C_{\alpha 2} + A_i A_{x+y+z} C_\beta + A_{x+y+z}^2 C_\beta,$

$\quad A_i A_{x+y+z} C_{\alpha 1} + A_{x+y+z}^2 C_{\alpha 1} + A_i A_{x+y+z} C_{\alpha 2} + B_\beta C_{\alpha 2} + B_\alpha C_\beta,$

$\quad A_i^3 A_{x+y+z}^2 + A_i^2 A_{x+y+z}^3 + A_i A_{x+y+z}^4 + A_{x+y+z}^5 + A_i^2 A_{x+y+z} B_\alpha + A_{x+y+z} B_\alpha^2 + A_i B_\alpha B_\beta + A_i A_{x+y+z}^2 B_{xy+xz+yz}$

$\quad + A_{x+y+z}^3 B_{xy+xz+yz} + A_i A_{x+y+z} C_{\alpha 1} + A_{x+y+z}^2 C_{\alpha 1} + A_i A_{x+y+z} C_{\alpha 2} + B_\beta C_{\alpha 2} + A_{x+y+z}^2 C_\beta + B_\beta C_\beta,$

$\quad A_i A_{x+y+z}^2 B_\alpha + A_{x+y+z} B_\alpha^2 + A_{x+y+z} B_\alpha B_\beta + A_i A_{x+y+z} C_{\alpha 1}$

$\quad + A_{x+y+z}^2 C_{\alpha 1} + B_\beta C_{\alpha 2} + A_i^2 C_\beta + A_i A_{x+y+z} C_\beta + B_{xy+xz+yz} C_\beta,$

$\quad A_i^2 A_{x+y+z}^3 + A_i A_{x+y+z}^4 + A_i^2 A_{x+y+z} B_\alpha + A_i A_{x+y+z}^2 B_\alpha + A_i A_{x+y+z}^2 B_\beta + A_{x+y+z}^3 B_\beta + A_{x+y+z} B_\alpha B_\beta$

$\quad + A_i A_{x+y+z} C_{\alpha 1} + B_\beta C_{\alpha 1} + B_\beta C_{\alpha 2} + A_i A_{x+y+z} C_\beta + B_\alpha C_{xyz},$

$\quad (B_\beta + B_{xy+xz+yz}) C_{xyz},$

$\quad A_i^4 A_{x+y+z} + A_i^3 A_{x+y+z}^2 + A_i A_{x+y+z}^2 B_\beta + A_{x+y+z}^3 B_\beta + A_{x+y+z} B_\alpha B_\beta + A_i^2 A_{x+y+z} B_{xy+xz+yz} + B_\beta C_{\alpha 1}$

$\quad + A_i A_{x+y+z} C_{\alpha 2} + B_\beta C_{\alpha 2} + A_i A_{x+y+z} C_\beta + B_{xy+xz+yz} C_{xyz},$ $\qquad$ (741b)

$\mathcal{R}_6:$ $B_\alpha^3 + C_{\alpha 1}^2 + C_{\alpha 1} C_{\alpha 2} + C_{\alpha 2}^2, \quad (C_{\alpha 2} + C_\beta) C_{xyz},$

$\quad A_i^4 A_{x+y+z}^2 + A_i^3 A_{x+y+z}^3 + A_i^2 A_{x+y+z}^4 + A_i A_{x+y+z}^5 + A_i^2 A_{x+y+z}^2 B_\alpha + A_i A_{x+y+z}^3 B_\alpha + A_{x+y+z}^4 B_\alpha + A_i A_{x+y+z} B_\alpha^2$

$\quad + A_{x+y+z}^2 B_\alpha^2 + A_i A_{x+y+z} B_\alpha B_\beta + B_\alpha^2 B_\beta + A_i^2 A_{x+y+z}^2 B_{xy+xz+yz} + A_{x+y+z}^4 B_{xy+xz+yz} + A_i^2 A_{x+y+z} C_{\alpha 1}$

$\quad + A_i A_{x+y+z}^2 C_{\alpha 1} + A_{x+y+z} B_\beta C_{\alpha 1} + A_i A_{x+y+z}^2 C_\beta + A_{x+y+z}^3 C_\beta + C_{\alpha 2} C_\beta + C_{\alpha 1} C_{xyz},$

$\quad A_i^5 A_{x+y+z} + A_i^4 A_{x+y+z}^2 + A_i^3 A_{x+y+z}^3 + A_i^2 A_{x+y+z}^4 + A_i^3 A_{x+y+z} B_\alpha + A_i^2 A_{x+y+z}^2 B_\alpha + A_{x+y+z}^4 B_\alpha$

$\quad + A_i A_{x+y+z} B_\alpha B_\beta + B_\alpha^2 B_\beta + A_i^3 A_{x+y+z} B_{xy+xz+yz} + A_{x+y+z}^4 B_{xy+xz+yz} + A_i^2 A_{x+y+z} C_{\alpha 1}$

$\quad + A_{x+y+z} B_\alpha C_{\alpha 1} + A_i B_\beta C_{\alpha 1} + A_{x+y+z} B_\beta C_{\alpha 1} + A_i A_{x+y+z}^2 C_\beta + A_{x+y+z}^3 C_\beta + C_\beta^2 + C_{\alpha 1} C_{xyz},$

$\quad A_i^4 A_{x+y+z}^2 + A_i^2 A_{x+y+z}^4 + A_i^3 A_{x+y+z} B_\alpha + A_i^2 A_{x+y+z}^2 B_\alpha + A_i A_{x+y+z} B_\alpha^2 + A_i^2 A_{x+y+z}^2 B_\beta + A_{x+y+z}^4 B_\beta$

$\quad + A_i A_{x+y+z} B_\alpha B_\beta + A_{x+y+z}^2 B_\alpha B_\beta + B_\alpha^2 B_\beta + A_i^2 A_{x+y+z}^2 B_{xy+xz+yz} + A_{x+y+z}^4 B_{xy+xz+yz} + A_i^2 A_{x+y+z} C_{\alpha 1}$

$\quad + A_i A_{x+y+z}^2 C_{\alpha 1} + A_{x+y+z}^3 C_{\alpha 1} + A_{x+y+z} B_\alpha C_{\alpha 1} + A_i^2 A_{x+y+z} C_{\alpha 2} + A_{x+y+z} B_\alpha C_{\alpha 2} + A_i^2 A_{x+y+z} C_\beta$

$\quad + A_i A_{x+y+z}^2 C_\beta + C_{\alpha 1} C_\beta + C_{\alpha 1} C_{xyz} + C_\beta C_{xyz},$

$\quad A_i^2 A_{x+y+z}^4 + A_i A_{x+y+z}^5 + A_i A_{x+y+z}^3 B_\alpha + A_{x+y+z}^4 B_\alpha + A_{x+y+z}^2 B_\alpha^2 + A_i A_{x+y+z}^3 B_\beta + A_{x+y+z}^4 B_\beta + A_{x+y+z}^2 B_\alpha B_\beta$

$\quad + B_\alpha^2 B_\beta + A_{x+y+z}^4 B_{xy+xz+yz} + A_i^2 A_{x+y+z} C_{\alpha 1} + A_{x+y+z}^3 C_{\alpha 1} + A_i^2 A_{x+y+z} C_{\alpha 2} + A_i A_{x+y+z}^2 C_\beta + C_{\alpha 1} C_\beta + C_{xyz}^2.$ $\qquad$ (741c)

We have the following table regarding IWPs and group cohomology at degree 3.

| Wyckoff position | Little group Intl. | Little group Schönflies | Coordinates | LSM anomaly class | Topo. inv. |
|---|---|---|---|---|---|
| 1a | $m\overline{3}$ | $T_h$ | $(0,0,0)$ | $(A_i + A_{x+y+z})^3 + (A_i + A_{x+y+z})B_\alpha + A_i B_{xy+xz+yz} + C_{\alpha 1} + C_{\alpha 2} + C_{xyz}$ | $\varphi_1[I]$ |
| 1b | $m\overline{3}$ | $T_h$ | $(1/2,1/2,1/2)$ | $C_{xyz}$ | $\varphi_1[T_1 T_2 T_3 I]$ |
| 3c | $mmm$ | $D_{2h}$ | $(0,1/2,1/2), (1/2,0,1/2), (1/2,1/2,0)$ | $(A_i + A_{x+y+z})B_{xy+xz+yz}$ | $\varphi_2[T_1 T_2 C_2, T_1 C_2']$ |
| 3d | $mmm$ | $D_{2h}$ | $(1/2,0,0), (0,1/2,0), (0,0,1/2)$ | $A_{x+y+z}(A_i^2 + A_i A_{x+y+z} + A_{x+y+z}^2 + B_\alpha + B_{xy+xz+yz})$ | $\varphi_2[T_1 C_2, T_1 C_2']$ |

**No. 201: $Pn\overline{3}$**

This group is generated by three translations $T_{1,2,3}$ as given in Eqs. (88), a two-fold rotation $C_2$, a two-fold rotation $C_2'$, a three-fold rotation $C_3$, and an inversion $I$:

$$C_2 \colon (x,y,z) \to (-x+1/2, -y+1/2, z), \tag{742a}$$
$$C_2' \colon (x,y,z) \to (-x+1/2, y, -z+1/2), \tag{742b}$$
$$C_3 \colon (x,y,z) \to (z,x,y), \tag{742c}$$
$$I \colon (x,y,z) \to (-x,-y,-z). \tag{742d}$$

The $\mathbb{Z}_2$ cohomology ring is given by

$$\mathbb{Z}_2[A_i, A_{x+y+z}, B_\alpha, C_{\alpha 1}, C_{\alpha 2}]/\langle \mathcal{R}_3, \mathcal{R}_4, \mathcal{R}_6 \rangle \tag{743}$$

where the relations are

$$\mathcal{R}_3 \colon \quad A_i A_{x+y+z}(A_i + A_{x+y+z}), \quad A_i B_\alpha, \tag{744a}$$
$$\mathcal{R}_4 \colon \quad A_i C_{\alpha 1}, \quad A_i C_{\alpha 2}, \quad A_{x+y+z}(A_i^3 + A_{x+y+z}^3 + A_{x+y+z}B_\alpha + C_{\alpha 1} + C_{\alpha 2}), \tag{744b}$$
$$\mathcal{R}_6 \colon \quad B_\alpha^3 + C_{\alpha 1}^2 + C_{\alpha 1}C_{\alpha 2} + C_{\alpha 2}^2. \tag{744c}$$

We have the following table regarding IWPs and group cohomology at degree 3.

| Wyckoff position | Little group Intl. | Little group Schönflies | Coordinates | LSM anomaly class | Topo. inv. |
|---|---|---|---|---|---|
| 2a | 23 | $T$ | $(1/4,1/4,1/4), (3/4,3/4,3/4)$ | $A_i^2 A_{x+y+z} + A_{x+y+z}^3 + A_{x+y+z}B_\alpha + C_{\alpha 1} + C_{\alpha 2}$ | $\varphi_2[C_2, C_2']$ |
| 4b | $\overline{3}$ | $C_{3i}$ | $(0,0,0), (1/2,1/2,0), (1/2,0,1/2), (0,1/2,1/2)$ | $A_i^2(A_i + A_{x+y+z})$ | $\varphi_1[I]$ |
| 4c | $\overline{3}$ | $C_{3i}$ | $(1/2,1/2,1/2), (0,0,1/2), (0,1/2,0), (1/2,0,0)$ | $A_i^2 A_{x+y+z}$ | $\varphi_1[T_1 I]$ |
| 6d | 222 | $D_2$ | $(1/4,3/4,3/4), (3/4,1/4,3/4), (3/4,3/4,1/4), (3/4,1/4,1/4), (1/4,3/4,1/4), (1/4,1/4,3/4)$ | $A_{x+y+z}(A_i^2 + A_{x+y+z}^2 + B_\alpha)$ | $\varphi_2[C_2, T_3 C_2']$ |

**No. 202: $Fm\overline{3}$**

This group is generated by three translations $T_{1,2,3}$ as given in Eqs. (92), a two-fold rotation $C_2$, a two-fold rotation $C_2'$, a three-fold rotation $C_3$, and an inversion $I$:

$$C_2 \colon (x,y,z) \to (-x, -y, z), \tag{745a}$$
$$C_2' \colon (x,y,z) \to (-x, y, -z), \tag{745b}$$
$$C_3 \colon (x,y,z) \to (z,x,y), \tag{745c}$$
$$I \colon (x,y,z) \to (-x,-y,-z). \tag{745d}$$

The $\mathbb{Z}_2$ cohomology ring is given by

$$\mathbb{Z}_2[A_i, B_\alpha, B_\beta, B_{xy+xz+yz}, C_{\alpha 1}, C_{\alpha 2}, C_{\beta 1}, C_{\beta 2}, C_{xyz}]/\langle \mathcal{R}_4, \mathcal{R}_5, \mathcal{R}_6 \rangle \tag{746}$$

where the relations are

$$\mathscr{R}_4: \quad A_i C_{\beta 2}, \quad A_i^2 B_\beta + B_\alpha B_{xy+xz+yz} + A_i C_{\beta 1},$$

$$A_i^2 B_\beta + B_\alpha B_\beta + B_\beta^2 + A_i C_{\beta 1}, \quad B_\beta B_{xy+xz+yz} + A_i C_{\beta 1}, \quad A_i^2 B_\beta + A_i^2 B_{xy+xz+yz} + B_{xy+xz+yz}^2, \tag{747a}$$

$$\mathscr{R}_5: \quad A_i B_\alpha B_\beta + B_{xy+xz+yz} C_{\alpha 1}, \quad A_i B_\alpha B_\beta + B_{xy+xz+yz} C_{\alpha 2}, \quad A_i B_\alpha B_\beta + B_\beta C_{\alpha 2} + B_\alpha C_{\beta 1},$$

$$A_i^3 B_\beta + B_\beta C_{\alpha 2} + A_i^2 C_{\beta 1} + B_\beta C_{\beta 1}, \quad A_i^3 B_\beta + A_i B_\alpha B_\beta + B_{xy+xz+yz} C_{\beta 1}, \quad B_\beta C_{\alpha 1} + B_\beta C_{\alpha 2} + B_\alpha C_{\beta 2},$$

$$B_\beta (C_{\alpha 1} + C_{\alpha 2} + C_{\beta 2}), \quad B_{xy+xz+yz} C_{\beta 2}, \quad B_\beta (A_i^3 + A_i B_\alpha + C_{xyz}),$$

$$A_i^3 B_\beta + A_i^3 B_{xy+xz+yz} + A_i^2 C_{\beta 1} + B_{xy+xz+yz} C_{xyz}, \tag{747b}$$

$$\mathscr{R}_6: \quad B_\alpha^3 + C_{\alpha 1}^2 + C_{\alpha 1} C_{\alpha 2} + C_{\alpha 2}^2, \quad B_\alpha^2 B_\beta + A_i B_\beta C_{\alpha 1} + C_{\alpha 2} C_{\beta 1} + C_{\alpha 1} C_{\beta 2},$$

$$B_\alpha^2 B_\beta + A_i B_\beta C_{\alpha 1} + C_{\alpha 1} C_{\beta 1} + C_{\alpha 1} C_{\beta 2} + C_{\alpha 2} C_{\beta 2},$$

$$A_i^4 B_\beta + A_i^2 B_\alpha B_\beta + B_\alpha^2 B_\beta + A_i B_\beta C_{\alpha 1} + A_i^3 C_{\beta 1} + C_{\beta 1}^2 + C_{\alpha 1} C_{\beta 2},$$

$$B_\alpha^2 B_\beta + A_i B_\beta C_{\alpha 1} + C_{\alpha 1} C_{\beta 1} + C_{\alpha 1} C_{\beta 2} + C_{\beta 1} C_{\beta 2}, \quad A_i^2 B_\alpha B_\beta + A_i B_\beta C_{\alpha 1} + A_i^3 C_{\beta 1} + C_{\beta 1} C_{xyz},$$

$$B_\alpha^2 B_\beta + A_i B_\beta C_{\alpha 1} + C_{\alpha 1} C_{\beta 1} + C_{\beta 2}^2, \quad C_{\beta 2} C_{xyz}, \quad C_{xyz} (A_i^3 + A_i B_\alpha + C_{\alpha 1} + C_{\alpha 2} + C_{xyz}). \tag{747c}$$

We have the following table regarding IWPs and group cohomology at degree 3.

| Wyckoff position | Little group | | Coordinates $(0,0,0)+$ $(0,1/2,1/2)+$ $(1/2,0,1/2)+$ $(1/2,1/2,0)+$ | LSM anomaly class | Topo. inv. |
|---|---|---|---|---|---|
| | Intl. | Schönflies | | | |
| 4a | $m\bar{3}$ | $T_h$ | $(0,0,0)$ | $A_i^3 + A_i B_\alpha + A_i B_\beta + A_i B_{xy+xz+yz} + C_{\alpha 1} + C_{\alpha 2} + C_{\beta 2} + C_{xyz}$ | $\varphi_1[I]$ |
| 4b | $m\bar{3}$ | $T_h$ | $(1/2,1/2,1/2)$ | $C_{xyz}$ | $\varphi_1[T_1 T_2 T_3^{-1} I]$ |
| 8c | 23 | $T$ | $(1/4,1/4,1/4), (3/4,3/4,3/4)$ | $C_{\beta 2}$ | $\varphi_2[T_3 C_2, T_2 C_2']$ |
| 24d | $2/m$ | $C_{2h}$ | $(0,1/4,1/4), (0,3/4,1/4),$ $(1/4,0,1/4), (1/4,0,3/4),$ $(1/4,1/4,0), (3/4,1/4,0)$ | $A_i(B_\beta + B_{xy+xz+yz})$ | $\varphi_1[T_1 I]$ |

## No. 203: $Fd\bar{3}$

This group is generated by three translations $T_{1,2,3}$ as given in Eqs. (92), a two-fold rotation $C_2$, a two-fold rotation $C_2'$, a three-fold rotation $C_3$, and an inversion $I$:

$$C_2: (x,y,z) \to (-x+1/4, -y+1/4, z), \tag{748a}$$

$$C_2': (x,y,z) \to (-x+1/4, y, -z+1/4), \tag{748b}$$

$$C_3: (x,y,z) \to (z,x,y), \tag{748c}$$

$$I: (x,y,z) \to (-x,-y,-z). \tag{748d}$$

The $\mathbb{Z}_2$ cohomology ring is given by

$$\mathbb{Z}_2[A_i, B_\alpha, B_{xy+xz+yz}, C_{\alpha 1}, C_{\alpha 2}, C_\gamma]/\langle \mathscr{R}_3, \mathscr{R}_4, \mathscr{R}_5, \mathscr{R}_6 \rangle \tag{749}$$

where the relations are

$$\mathscr{R}_3: \quad A_i B_\alpha, \tag{750a}$$

$$\mathscr{R}_4: \quad A_i C_{\alpha 1}, \quad A_i C_{\alpha 2}, \quad A_i C_\gamma, \quad B_\alpha(B_\alpha + B_{xy+xz+yz}), \quad B_\alpha^2 + A_i^2 B_{xy+xz+yz} + B_{xy+xz+yz}^2, \tag{750b}$$

$$\mathscr{R}_5: \quad (B_\alpha + B_{xy+xz+yz})C_{\alpha 1}, \quad (B_\alpha + B_{xy+xz+yz})C_{\alpha 2}, \quad (B_\alpha + B_{xy+xz+yz})C_\gamma, \tag{750c}$$

$$\mathscr{R}_6: \quad B_\alpha^3 + C_{\alpha 1}^2 + C_{\alpha 1} C_{\alpha 2} + C_{\alpha 2}^2, \quad C_\gamma(C_{\alpha 1} + C_{\alpha 2} + C_\gamma). \tag{750d}$$

We have the following table regarding IWPs and group cohomology at degree 3.

| Wyckoff position | Little group | | Coordinates $(0,0,0)+$ $(0,1/2,1/2)+$ $(1/2,0,1/2)+$ $(1/2,1/2,0)+$ | LSM anomaly class | Topo. inv. |
|---|---|---|---|---|---|
| | Intl. | Schönflies | | | |
| 8a | 23 | $T$ | $(1/8,1/8,1/8), (7/8,7/8,7/8)$ | $C_\gamma$ | $\varphi_2[C_2, C_2']$ |
| 8b | 23 | $T$ | $(5/8,5/8,5/8), (3/8,3/8,3/8)$ | $C_{\alpha 1} + C_{\alpha 2} + C_\gamma$ | $\varphi_2[T_3 C_2, T_2 C_2']$ |
| 16c | $\bar{3}$ | $C_{3i}$ | $(0,0,0), (3/4,3/4,0),$ $(3/4,0,3/4), (0,3/4,3/4)$ | $A_i(A_i^2 + B_{xy+xz+yz})$ | $\varphi_1[I]$ |
| 16d | $\bar{3}$ | $C_{3i}$ | $(1/2,1/2,1/2), (1/4,1/4,1/2),$ $(1/4,1/2,1/4), (1/2,1/4,1/4)$ | $A_i B_{xy+xz+yz}$ | $\varphi_1[T_1 T_2 T_3^{-1} I]$ |

**No. 204:** $Im\bar{3}$

This group is generated by three translations $T_{1,2,3}$ as given in Eqs. (91), a two-fold rotation $C_2$, a two-fold rotation $C_2'$, a three-fold rotation $C_3$, and an inversion $I$:

$$C_2 : (x, y, z) \rightarrow (-x, -y, z), \tag{751a}$$
$$C_2' : (x, y, z) \rightarrow (-x, y, -z), \tag{751b}$$
$$C_3 : (x, y, z) \rightarrow (z, x, y), \tag{751c}$$
$$I : (x, y, z) \rightarrow (-x, -y, -z). \tag{751d}$$

The $\mathbb{Z}_2$ cohomology ring is given by

$$\mathbb{Z}_2[A_i, A_{x+y+z}, B_\alpha, B_\beta, C_{\alpha 1}, C_{\alpha 2}, C_\gamma, C_{xyz}, D_\delta]/\langle \mathcal{R}_2, \mathcal{R}_3, \mathcal{R}_4, \mathcal{R}_5, \mathcal{R}_6 \rangle \tag{752}$$

where the relations are

$$\mathcal{R}_2 : \quad A_{x+y+z}(A_i + A_{x+y+z}), \tag{753a}$$
$$\mathcal{R}_3 : \quad A_{x+y+z}B_\alpha, \quad A_{x+y+z}(A_i^2 + B_\beta), \tag{753b}$$
$$\mathcal{R}_4 : \quad A_{x+y+z}C_{\alpha 1}, \quad A_{x+y+z}C_{\alpha 2}, \quad A_{x+y+z}C_\gamma, \quad A_{x+y+z}C_{xyz}, \tag{753c}$$
$$\mathcal{R}_5 : \quad A_{x+y+z}D_\delta, \quad A_i B_\alpha B_\beta + A_i^2 C_\gamma + B_\alpha C_\gamma + B_\beta C_\gamma + A_i^2 C_{xyz} + B_\alpha C_{xyz} + A_i D_\delta,$$
$$A_i^4 A_{x+y+z} + A_i^3 B_\beta + A_i B_\alpha B_\beta + B_\beta C_{\alpha 1} + B_\beta C_{\alpha 2} + B_\beta C_{xyz}, \tag{753d}$$
$$\mathcal{R}_6 : \quad A_i^2 B_\alpha B_\beta + B_\alpha^2 B_\beta + B_\alpha B_\beta^2 + A_i B_\beta C_{\alpha 1} + A_i B_\beta C_{\alpha 2} + A_i^3 C_\gamma + A_i B_\beta C_\gamma + C_{\alpha 1} C_\gamma + A_i^3 C_{xyz} + A_i^2 D_\delta + B_\alpha D_\delta,$$
$$A_i^4 B_\beta + A_i^2 B_\alpha B_\beta + B_\alpha^2 B_\beta + B_\beta^3 + A_i B_\beta C_{\alpha 2} + A_i^3 C_\gamma + A_i B_\alpha C_\gamma + A_i^3 C_{xyz} + C_{\alpha 1} C_{xyz} + A_i^2 D_\delta + B_\alpha D_\delta + B_\beta D_\delta,$$
$$B_\alpha^3 + C_{\alpha 1}^2 + C_{\alpha 1} C_{\alpha 2} + C_{\alpha 2}^2, \quad A_i^2 B_\alpha B_\beta + A_i B_\beta C_{\alpha 1} + A_i^3 C_\gamma + A_i B_\beta C_\gamma + C_{\alpha 2} C_\gamma + A_i^3 C_{xyz} + A_i^2 D_\delta + B_\alpha D_\delta,$$
$$A_i^5 A_{x+y+z} + A_i^4 B_\beta + A_i B_\beta C_{\alpha 1} + A_i^3 C_\gamma + A_i B_\alpha C_\gamma + A_i^3 C_{xyz} + C_{\alpha 2} C_{xyz} + A_i^2 D_\delta + B_\alpha D_\delta + B_\beta D_\delta,$$
$$A_i^5 A_{x+y+z} + B_\alpha^2 B_\beta + A_i^2 B_\beta^2 + A_i^3 C_\gamma + A_i B_\alpha C_\gamma + C_\gamma^2 + A_i^3 C_{xyz} + A_i^2 D_\delta,$$
$$B_\alpha^2 B_\beta + B_\alpha B_\beta^2 + A_i B_\beta C_{\alpha 2} + A_i^3 C_\gamma + A_i B_\alpha C_\gamma + C_\gamma C_{xyz},$$
$$A_i^5 A_{x+y+z} + B_\alpha^2 B_\beta + B_\beta^3 + A_i B_\beta C_{\alpha 1} + A_i B_\beta C_{\alpha 2} + A_i^3 C_\gamma + A_i B_\alpha C_\gamma + A_i B_\beta C_\gamma + C_{xyz}^2 + A_i^2 D_\delta, \tag{753e}$$
$$\mathcal{R}_7 : \quad A_i^5 B_\beta + A_i B_\alpha^2 B_\beta + A_i B_\alpha B_\beta^2 + A_i B_\beta^3 + A_i^2 B_\beta C_{\alpha 1} + B_\alpha B_\beta C_{\alpha 1} + B_\beta^2 C_{\alpha 1} + B_\alpha B_\beta C_{\alpha 2} + B_\beta^2 C_{\alpha 2}$$
$$+ A_i^2 B_\alpha C_\gamma + B_\alpha^2 C_\gamma + A_i^2 B_\beta C_\gamma + A_i B_\beta D_\delta + C_{\alpha 1} D_\delta,$$
$$A_i^6 A_{x+y+z} + A_i^5 B_\beta + A_i^3 B_\alpha B_\beta + B_\alpha B_\beta C_{\alpha 1} + B_\beta^2 C_{\alpha 1} + A_i^2 B_\alpha C_\gamma + B_\alpha^2 C_\gamma + A_i^2 B_\beta C_\gamma + A_i B_\beta D_\delta + C_{\alpha 2} D_\delta,$$
$$A_i^6 A_{x+y+z} + A_i^5 B_\beta + A_i^3 B_\beta^2 + A_i B_\beta^3 + A_i^2 B_\beta C_{\alpha 1} + B_\alpha B_\beta C_{\alpha 2} + A_i^4 C_\gamma + A_i^2 B_\alpha C_\gamma + A_i B_\alpha D_\delta + A_i B_\beta D_\delta + C_\gamma D_\delta,$$
$$A_i^6 A_{x+y+z} + A_i^3 B_\alpha B_\beta + A_i B_\alpha^2 B_\beta + A_i B_\alpha B_\beta^2 + A_i B_\beta^3 + A_i^2 B_\beta C_{\alpha 1} + B_\alpha B_\beta C_{\alpha 2} + B_\beta^2 C_{\alpha 2}$$
$$+ A_i^3 D_\delta + A_i B_\alpha D_\delta + C_{xyz} D_\delta, \tag{753f}$$
$$\mathcal{R}_8 : \quad A_i^7 A_{x+y+z} + A_i^6 B_\beta + B_\alpha^3 B_\beta + B_\alpha^2 B_\beta^2 + A_i^2 B_\beta^3 + B_\beta^4 + A_i B_\beta^2 C_{\alpha 1} + A_i B_\beta^2 C_{\alpha 2} + A_i^5 C_\gamma + A_i B_\alpha^2 C_\gamma + A_i^3 B_\beta C_\gamma$$
$$+ A_i B_\alpha B_\beta C_\gamma + A_i^4 D_\delta + A_i^2 B_\alpha D_\delta + A_i^2 B_\beta D_\delta + B_\alpha B_\beta D_\delta + B_\beta^2 D_\delta + D_\delta^2. \tag{753g}$$

We have the following table regarding IWPs and group cohomology at degree 3.

| Wyckoff position | Little group | | Coordinates $(0,0,0)+$ $(1/2,1/2,1/2)+$ | LSM anomaly class | Topo. inv. |
|---|---|---|---|---|---|
| | Intl. | Schönflies | | | |
| 2a | $m\bar{3}$ | $T_h$ | $(0,0,0)$ | $A_i^3 + A_i^2 A_{x+y+z} + A_i B_\alpha + C_{\alpha 1} + C_{\alpha 2} + C_{xyz}$ | $\varphi_1[I]$ |
| 6b | $mmm$ | $D_{2h}$ | $(0,1/2,1/2), (1/2,0,1/2),$ $(1/2,1/2,0)$ | $C_{xyz}$ | $\varphi_2[C_2, T_1 T_2 C_2']$ |
| 8c | $\bar{3}$ | $C_{3i}$ | $(1/4,1/4,1/4), (3/4,3/4,1/4),$ $(3/4,1/4,3/4), (1/4,3/4,3/4)$ | $A_i^2 A_{x+y+z}$ | $\varphi_1[T_1 T_2 T_3 I]$ |

**No. 205: $Pa\bar{3}$**

This group is generated by three translations $T_{1,2,3}$ as given in Eqs. (88), a two-fold screw $S_2$, a two-fold screw $S_2'$, a three-fold rotation $C_3$, and an inversion $I$:

$$S_2: (x,y,z) \to (-x+1/2, -y, z+1/2), \tag{754a}$$
$$S_2': (x,y,z) \to (-x, y+1/2, -z+1/2), \tag{754b}$$
$$C_3: (x,y,z) \to (z,x,y), \tag{754c}$$
$$I: (x,y,z) \to (-x,-y,-z). \tag{754d}$$

The $\mathbb{Z}_2$ cohomology ring is given by

$$\mathbb{Z}_2[A_i, C_\beta]/\langle \mathscr{R}_6 \rangle \tag{755}$$

where the relations are

$$\mathscr{R}_6: \quad C_\beta(A_i^3 + C_\beta). \tag{756a}$$

We have the following table regarding IWPs and group cohomology at degree 3.

| Wyckoff position | Little group Intl. | Little group Schönflies | Coordinates | LSM anomaly class | Topo. inv. |
|---|---|---|---|---|---|
| 4a | $\bar{3}$ | $C_{3i}$ | $(0,0,0)$, $(1/2,0,1/2)$, $(0,1/2,1/2)$, $(1/2,1/2,0)$ | $C_\beta$ | $\varphi_1[I]$ |
| 4b | $\bar{3}$ | $C_{3i}$ | $(1/2,1/2,1/2)$, $(0,1/2,0)$, $(1/2,0,0)$, $(0,0,1/2)$ | $A_i^3 + C_\beta$ | $\varphi_1[T_3 I]$ |

**No. 206: $Ia\bar{3}$**

This group is generated by three translations $T_{1,2,3}$ as given in Eqs. (91), a two-fold rotation $C_2$, a two-fold rotation $C_2'$, a three-fold rotation $C_3$, and an inversion $I$:

$$C_2: (x,y,z) \to (-x, -y+1/2, z), \tag{757a}$$
$$C_2': (x,y,z) \to (-x+1/2, y, -z), \tag{757b}$$
$$C_3: (x,y,z) \to (z,x,y), \tag{757c}$$
$$I: (x,y,z) \to (-x,-y,-z). \tag{757d}$$

The $\mathbb{Z}_2$ cohomology ring is given by

$$\mathbb{Z}_2[A_i, A_{x+y+z}]/\langle \mathscr{R}_4 \rangle \tag{758}$$

where the relations are

$$\mathscr{R}_4: \quad A_i^2 A_{x+y+z}(A_i + A_{x+y+z}). \tag{759a}$$

We have the following table regarding IWPs and group cohomology at degree 3.

| Wyckoff position | Little group Intl. | Little group Schönflies | Coordinates $(0,0,0)+$ $(1/2,1/2,1/2)+$ | LSM anomaly class | Topo. inv. |
|---|---|---|---|---|---|
| 8a | $\bar{3}$ | $C_{3i}$ | $(0,0,0)$, $(1/2,0,1/2)$, $(0,1/2,1/2)$, $(1/2,1/2,0)$ | $A_i^2(A_i + A_{x+y+z})$ | $\varphi_1[I]$ |
| 8b | $\bar{3}$ | $C_{3i}$ | $(1/4,1/4,1/4)$, $(1/4,3/4,3/4)$, $(3/4,3/4,1/4)$, $(3/4,1/4,3/4)$ | $A_i^2 A_{x+y+z}$ | $\varphi_1[T_1 T_2 T_3 I]$ |
| 24d | 2 | $C_2$ | $(x,0,1/4)$, $(-x+1/2,0,3/4)$, $(1/4,x,0)$, $(3/4,-x+1/2,0)$, $(0,1/4,x)$, $(0,3/4,-x+1/2)$, $(-x,0,3/4)$, $(x+1/2,0,1/4)$, $(3/4,-x,0)$, $(1/4,x+1/2,0)$, $(0,3/4,-x)$, $(0,1/4,x+1/2)$ | $A_i A_{x+y+z}(A_i + A_{x+y+z})$ | $\varphi_2[T_1 C_2' I, C_2]$ |

**No. 207:** $P432$

This group is generated by three translations $T_{1,2,3}$ as given in Eqs. (88), a two-fold rotation $C_2$, a two-fold rotation $C_2'$, a three-fold rotation $C_3$, and a two-fold rotation $C_2''$:

$$C_2\colon (x,y,z) \to (-x,-y,z), \tag{760a}$$
$$C_2'\colon (x,y,z) \to (-x,y,-z), \tag{760b}$$
$$C_3\colon (x,y,z) \to (z,x,y), \tag{760c}$$
$$C_2''\colon (x,y,z) \to (-y,-x,-z). \tag{760d}$$

The $\mathbb{Z}_2$ cohomology ring is given by

$$\mathbb{Z}_2[A_{c''}, A_{x+y+z}, B_\alpha, B_{xy+xz+yz}, C_\alpha, C_{xyz}]/\langle \mathcal{R}_3, \mathcal{R}_4, \mathcal{R}_5, \mathcal{R}_6 \rangle \tag{761}$$

where the relations are

$$\mathcal{R}_3\colon \quad A_{c''}A_{x+y+z}(A_{c''} + A_{x+y+z}), \tag{762a}$$

$$\mathcal{R}_4\colon \quad A_{c''}C_\alpha, \quad A_{x+y+z}(A_{c''}^3 + A_{x+y+z}^3 + A_{c''}B_\alpha + A_{x+y+z}B_\alpha + C_\alpha), \quad A_{c''}(A_{x+y+z}B_{xy+xz+yz} + C_{xyz}),$$
$$A_{x+y+z}(A_{c''}B_{xy+xz+yz} + C_{xyz}), \quad B_{xy+xz+yz}(A_{c''}A_{x+y+z} + A_{x+y+z}^2 + B_\alpha + B_{xy+xz+yz}), \tag{762b}$$

$$\mathcal{R}_5\colon \quad A_{c''}^2 A_{x+y+z}B_{xy+xz+yz} + A_{x+y+z}^3 B_{xy+xz+yz} + A_{x+y+z}B_\alpha B_{xy+xz+yz} + B_{xy+xz+yz}C_\alpha + B_\alpha C_{xyz},$$
$$B_{xy+xz+yz}(A_{c''}^2 A_{x+y+z} + A_{x+y+z}^3 + A_{x+y+z}B_\alpha + C_\alpha + C_{xyz}), \tag{762c}$$

$$\mathcal{R}_6\colon \quad A_{c''}A_{x+y+z}B_\alpha B_{xy+xz+yz} + C_\alpha C_{xyz} + C_{xyz}^2. \tag{762d}$$

We have the following table regarding IWPs and group cohomology at degree 3.

| Wyckoff position | Little group Intl. | Little group Schönflies | Coordinates | LSM anomaly class | Topo. inv. |
|---|---|---|---|---|---|
| 1a | 432 | $O$ | $(0,0,0)$ | $A_{c''}^2 A_{x+y+z} + A_{x+y+z}^3 + A_{c''}B_\alpha + A_{x+y+z}B_\alpha + A_{c''}B_{xy+xz+yz} + C_\alpha + C_{xyz}$ | $\varphi_2[C_2, C_2']$ |
| 1b | 432 | $O$ | $(1/2,1/2,1/2)$ | $C_{xyz}$ | $\varphi_2[T_1T_2C_2, T_1T_3C_2']$ |
| 3c | 422 | $D_4$ | $(0,1/2,1/2), (1/2,0,1/2), (1/2,1/2,0)$ | $(A_{c''} + A_{x+y+z})B_{xy+xz+yz}$ | $\varphi_2[T_1T_2C_2, T_1C_2']$ |
| 3d | 422 | $D_4$ | $(1/2,0,0), (0,1/2,0), (0,0,1/2)$ | $A_{x+y+z}(A_{c''}^2 + A_{x+y+z}^2 + B_\alpha + B_{xy+xz+yz})$ | $\varphi_2[T_1C_2, T_1C_2']$ |

**No. 208:** $P4_232$

This group is generated by three translations $T_{1,2,3}$ as given in Eqs. (88), a two-fold rotation $C_2$, a two-fold rotation $C_2'$, a three-fold rotation $C_3$, and a two-fold rotation $C_2''$:

$$C_2\colon (x,y,z) \to (-x,-y,z), \tag{763a}$$
$$C_2'\colon (x,y,z) \to (-x,y,-z), \tag{763b}$$
$$C_3\colon (x,y,z) \to (z,x,y), \tag{763c}$$
$$C_2''\colon (x,y,z) \to (-y+1/2,-x+1/2,-z+1/2). \tag{763d}$$

The $\mathbb{Z}_2$ cohomology ring is given by

$$\mathbb{Z}_2[A_{c''}, A_{x+y+z}, B_\alpha, B_{xy+xz+yz}, C_\alpha, C_\beta]/\langle \mathcal{R}_3, \mathcal{R}_4, \mathcal{R}_5, \mathcal{R}_6 \rangle \tag{764}$$

where the relations are

$$\mathcal{R}_3: \quad A_{c''}(A_{c''}A_{x+y+z} + A_{x+y+z}^2 + B_\alpha), \tag{765a}$$

$$\mathcal{R}_4: \quad A_{c''}C_\alpha, \quad A_{x+y+z}(A_{c''}A_{x+y+z}^2 + A_{x+y+z}^3 + A_{x+y+z}B_\alpha + C_\alpha), \quad A_{c''}(A_{c''}^2 A_{x+y+z} + A_{x+y+z}^3 + C_\beta),$$

$$A_{c''}^2 A_{x+y+z}^2 + A_{x+y+z}^4 + B_\alpha^2 + A_{c''}A_{x+y+z}B_{xy+xz+yz} + A_{x+y+z}^2 B_{xy+xz+yz} + B_{xy+xz+yz}^2,$$

$$A_{c''}A_{x+y+z}^3 + A_{x+y+z}^4 + A_{x+y+z}^2 B_\alpha + B_\alpha^2 + A_{c''}A_{x+y+z}B_{xy+xz+yz} + A_{x+y+z}^2 B_{xy+xz+yz} + B_\alpha B_{xy+xz+yz}$$

$$+ A_{x+y+z}C_\beta, \tag{765b}$$

$$\mathcal{R}_5: \quad A_{c''}^2 A_{x+y+z}^3 + A_{x+y+z}^5 + B_\alpha C_\alpha + B_{xy+xz+yz}C_\alpha + A_{x+y+z}^2 C_\beta,$$

$$A_{c''}^3 A_{x+y+z}^2 + A_{c''}^2 A_{x+y+z}^3 + A_{x+y+z}^3 B_\alpha + A_{x+y+z}B_\alpha^2 + A_{c''}^2 A_{x+y+z}B_{xy+xz+yz} + A_{x+y+z}^3 B_{xy+xz+yz} + A_{x+y+z}^2 C_\beta$$

$$+ B_\alpha C_\beta + B_{xy+xz+yz}C_\beta, \tag{765c}$$

$$\mathcal{R}_6: \quad A_{c''}^4 A_{x+y+z}^2 + A_{c''}^3 A_{x+y+z}^3 + A_{c''}^2 A_{x+y+z}^4 + A_{c''}A_{x+y+z}^5 + B_\alpha^3 + C_\alpha^2 + C_\alpha C_\beta + C_\beta^2. \tag{765d}$$

We have the following table regarding IWPs and group cohomology at degree 3.

| Wyckoff position | Little group Intl. | Schönflies | Coordinates | LSM anomaly class | Topo. inv. |
|---|---|---|---|---|---|
| 2a | 23 | $T$ | $(0,0,0)$, $(1/2,1/2,1/2)$ | $A_{c''}A_{x+y+z}^2 + A_{x+y+z}^3 + A_{x+y+z}B_\alpha + C_\alpha$ | $\varphi_2[C_2, C_2']$ |
| 4b | 32 | $D_3$ | $(1/4,1/4,1/4)$, $(3/4,3/4,1/4)$, $(3/4,1/4,3/4)$, $(1/4,3/4,3/4)$ | N/A | N/A |
| 4c | 32 | $D_3$ | $(3/4,3/4,3/4)$, $(1/4,1/4,3/4)$, $(1/4,3/4,1/4)$, $(3/4,1/4,1/4)$ | N/A | N/A |
| 6d | 222 | $D_2$ | $(0,1/2,1/2)$, $(1/2,0,1/2)$, $(1/2,1/2,0)$, $(0,1/2,0)$, $(1/2,0,0)$, $(0,0,1/2)$ | $A_{x+y+z}(A_{c''}A_{x+y+z} + A_{x+y+z}^2 + B_\alpha)$ | $\varphi_2[T_1 C_2, T_1 C_2']$ |
| 6e | 222 | $D_2$ | $(1/4,0,1/2)$, $(3/4,0,1/2)$, $(1/2,1/4,0)$, $(1/2,3/4,0)$, $(0,1/2,1/4)$, $(0,1/2,3/4)$ | $A_{c''}(A_{c''}A_{x+y+z} + A_{x+y+z}^2 + B_{xy+xz+yz})$ | $\varphi_2[T_2 C_2, C_2'']$ |
| 6f | 222 | $D_2$ | $(1/4,1/2,0)$, $(3/4,1/2,0)$, $(0,1/4,1/2)$, $(0,3/4,1/2)$, $(1/2,0,1/4)$, $(1/2,0,3/4)$ | $A_{c''}B_{xy+xz+yz}$ | $\varphi_2[T_1 C_2, C_2'']$ |

**No. 209:** $F432$

This group is generated by three translations $T_{1,2,3}$ as given in Eqs. (92), a two-fold rotation $C_2$, a two-fold rotation $C_2'$, a three-fold rotation $C_3$, and a two-fold rotation $C_2''$:

$$C_2: (x,y,z) \to (-x,-y,z), \tag{766a}$$

$$C_2': (x,y,z) \to (-x,y,-z), \tag{766b}$$

$$C_3: (x,y,z) \to (z,x,y), \tag{766c}$$

$$C_2'': (x,y,z) \to (-y,-x,-z). \tag{766d}$$

The $\mathbb{Z}_2$ cohomology ring is given by

$$\mathbb{Z}_2[A_{c''}, B_\alpha, B_\beta, B_{xy+xz+yz}, C_\alpha, C_{\gamma 1}, C_{\gamma 2}, C_{xyz}]/\langle \mathcal{R}_4, \mathcal{R}_5, \mathcal{R}_6 \rangle \tag{767}$$

where the relations are

$\mathcal{R}_4:$  $A_{c''}C_\alpha,$  $A_{c''}(C_{\gamma1} + C_{\gamma2}),$  $A_{c''}^2 B_\beta + B_\alpha B_\beta + B_\alpha B_{xy+xz+yz} + A_{c''}C_{\gamma1},$  $B_\alpha B_\beta + B_\beta^2 + A_{c''}C_{\gamma1},$

$A_{c''}^2 B_\alpha + B_\alpha B_\beta + A_{c''}^2 B_{xy+xz+yz} + B_\beta B_{xy+xz+yz} + A_{c''}C_{\gamma1} + A_{c''}C_{xyz},$  $A_{c''}^2 B_\beta + B_\alpha B_\beta + B_{xy+xz+yz}^2 + A_{c''}C_{\gamma1},$

$$\tag{768a}$$

$\mathcal{R}_5:$  $(B_\beta + B_{xy+xz+yz})C_\alpha,$  $A_{c''}^3 B_\alpha + A_{c''}B_\alpha B_\beta + A_{c''}^3 B_{xy+xz+yz} + B_\beta C_{\gamma1} + B_{xy+xz+yz}C_{\gamma1} + A_{c''}^2 C_{xyz},$

$B_\beta C_\alpha + B_\alpha C_{\gamma1} + B_\alpha C_{\gamma2},$  $B_\beta(C_\alpha + C_{\gamma1} + C_{\gamma2}),$

$A_{c''}^3 B_\alpha + A_{c''}B_\alpha B_\beta + A_{c''}^3 B_{xy+xz+yz} + B_\beta C_\alpha + B_\beta C_{\gamma1} + B_{xy+xz+yz}C_{\gamma2} + A_{c''}^2 C_{xyz},$

$A_{c''}B_\alpha^2 + A_{c''}^3 B_\beta + B_\beta C_{\gamma1} + B_\alpha C_{xyz},$  $B_\beta(C_{\gamma1} + C_{xyz}),$

$A_{c''}^3 B_\alpha + A_{c''}B_\alpha B_\beta + A_{c''}^3 B_{xy+xz+yz} + B_\beta C_{\gamma1} + A_{c''}^2 C_{xyz} + B_{xy+xz+yz}C_{xyz},$

$$\tag{768b}$$

$\mathcal{R}_6:$  $A_{c''}^2 B_\alpha B_\beta + B_\alpha^2 B_\beta + A_{c''}^3 C_{\gamma1} + C_{\gamma1}^2 + C_\alpha C_{\gamma2},$  $A_{c''}^2 B_\alpha B_\beta + B_\alpha^2 B_\beta + A_{c''}^3 C_{\gamma1} + C_\alpha C_{\gamma2} + C_{\gamma1}C_{\gamma2} + C_\alpha C_{xyz},$

$A_{c''}^2 B_\alpha B_\beta + B_\alpha^2 B_\beta + A_{c''}^3 C_{\gamma1} + C_\alpha C_{\gamma1} + C_\alpha C_{\gamma2} + C_\alpha C_{xyz} + C_{\gamma1}C_{xyz},$

$A_{c''}^2 B_\alpha B_\beta + B_\alpha^2 B_\beta + A_{c''}^3 C_{\gamma1} + C_\alpha C_{\gamma1} + C_{\gamma2}^2,$  $A_{c''}^2 B_\alpha B_\beta + B_\alpha^2 B_\beta + A_{c''}^3 C_{\gamma1} + C_\alpha C_{\gamma1} + C_\alpha C_{\gamma2} + C_{\gamma2}C_{xyz},$

$A_{c''}^2 B_\alpha^2 + A_{c''}^4 B_\beta + A_{c''}^2 B_\alpha B_\beta + B_\alpha^2 B_\beta + A_{c''}^3 C_{\gamma1} + A_{c''}B_\alpha C_{\gamma1} + A_{c''}B_\beta C_{\gamma1} + C_\alpha C_{\gamma1} + C_\alpha C_{\gamma2} + C_\alpha C_{xyz} + C_{xyz}^2.$

$$\tag{768c}$$

We have the following table regarding IWPs and group cohomology at degree 3.

| Wyckoff position | Little group | | Coordinates $(0,0,0)+ \ (0,1/2,1/2)+$ $(1/2,0,1/2)+ \ (1/2,1/2,0)+$ | LSM anomaly class | Topo. inv. |
|---|---|---|---|---|---|
| | Intl. | Schönflies | | | |
| 4a | 432 | $O$ | $(0,0,0)$ | $C_\alpha + C_{\gamma2} + C_{xyz}$ | $\varphi_2[C_2, C_2']$ |
| 4b | 432 | $O$ | $(1/2,1/2,1/2)$ | $A_{c''}B_\alpha + A_{c''}B_{xy+xz+yz} + C_{\gamma1} + C_{xyz}$ | $\varphi_2[C_2, T_1T_2T_3^{-1}C_2']$ |
| 8c | 23 | $T$ | $(1/4,1/4,1/4), (1/4,1/4,3/4)$ | $C_{\gamma1} + C_{\gamma2}$ | $\varphi_2[T_3C_2, T_2C_2']$ |
| 24d | 222 | $D_2$ | $(0,1/4,1/4), (0,3/4,1/4),$ $(1/4,0,1/4), (1/4,0,3/4),$ $(1/4,1/4,0), (3/4,1/4,0)$ | $A_{c''}B_{xy+xz+yz}$ | $\varphi_2[T_3C_2, T_3C_2'']$ |

## No. 210: $F4_132$

This group is generated by three translations $T_{1,2,3}$ as given in Eqs. (92), a two-fold rotation $C_2$, a two-fold rotation $C_2'$, a three-fold rotation $C_3$, and a two-fold rotation $C_2''$:

$$C_2: (x,y,z) \rightarrow (-x,-y,z), \tag{769a}$$

$$C_2': (x,y,z) \rightarrow (-x,y,-z), \tag{769b}$$

$$C_3: (x,y,z) \rightarrow (z,x,y), \tag{769c}$$

$$C_2'': (x,y,z) \rightarrow (-y+1/4, -x+1/4, -z+1/4). \tag{769d}$$

The $\mathbb{Z}_2$ cohomology ring is given by

$$\mathbb{Z}_2[A_{c''}, B_\alpha, B_{xy+xz+yz}, C_\alpha, C_\beta, C_\gamma]/\langle \mathcal{R}_3, \mathcal{R}_4, \mathcal{R}_5, \mathcal{R}_6 \rangle \tag{770}$$

where the relations are

$$\mathcal{R}_3: \quad A_{c''}B_\alpha, \tag{771a}$$

$$\mathcal{R}_4: \quad A_{c''}C_\alpha, \quad A_{c''}C_\beta, \quad A_{c''}C_\gamma, \quad B_\alpha B_{xy+xz+yz}, \quad B_{xy+xz+yz}^2, \tag{771b}$$

$$\mathcal{R}_5: \quad B_{xy+xz+yz}C_\alpha, \quad B_{xy+xz+yz}C_\beta, \quad B_{xy+xz+yz}C_\gamma, \tag{771c}$$

$$\mathcal{R}_6: \quad B_\alpha^3 + C_\alpha^2 + C_\alpha C_\beta + C_\beta^2, \quad C_\gamma(C_\alpha + C_\gamma). \tag{771d}$$

We have the following table regarding IWPs and group cohomology at degree 3.

| Wyckoff position | Little group | | Coordinates $(0,0,0)+$ $(0,1/2,1/2)+$ $(1/2,0,1/2)+$ $(1/2,1/2,0)+$ | LSM anomaly class | Topo. inv. |
|---|---|---|---|---|---|
| | Intl. | Schönflies | | | |
| 8a | 23 | $T$ | $(0,0,0), (3/4,1/4,3/4)$ | $C_\alpha + C_\gamma$ | $\varphi_2[C_2, C_2']$ |
| 8b | 23 | $T$ | $(1/2,1/2,1/2), (1/4,3/4,1/4)$ | $C_\gamma$ | $\varphi_2[C_2, T_1 T_2 T_3^{-1} C_2']$ |
| 16c | 32 | $D_3$ | $(1/8,1/8,1/8), (7/8,3/8,5/8),$ $(3/8,5/8,7/8), (5/8,7/8,3/8)$ | $A_{c''} B_{xy+xz+yz}$ | $\varphi_2[T_1 T_2^{-1}, C_2'']$ |
| 16d | 32 | $D_3$ | $(5/8,5/8,5/8), (3/8,7/8,1/8),$ $(7/8,1/8,3/8), (1/8,3/8,7/8)$ | Same as 16c | Same as 16c |

## No. 211: $I432$

This group is generated by three translations $T_{1,2,3}$ as given in Eqs. (91), a two-fold rotation $C_2$, a two-fold rotation $C_2'$, a three-fold rotation $C_3$, and a two-fold rotation $C_2''$:

$$C_2: (x,y,z) \to (-x,-y,z), \tag{772a}$$
$$C_2': (x,y,z) \to (-x,y,-z), \tag{772b}$$
$$C_3: (x,y,z) \to (z,x,y), \tag{772c}$$
$$C_2'': (x,y,z) \to (-y,-x,-z). \tag{772d}$$

The $\mathbb{Z}_2$ cohomology ring is given by

$$\mathbb{Z}_2[A_{c''}, A_{x+y+z}, B_\alpha, B_\beta, C_\alpha, C_\gamma, C_{xyz}]/\langle \mathscr{R}_2, \mathscr{R}_3, \mathscr{R}_4, \mathscr{R}_5, \mathscr{R}_6 \rangle \tag{773}$$

where the relations are

$$\mathscr{R}_2: \quad A_{x+y+z}(A_{c''} + A_{x+y+z}), \tag{774a}$$
$$\mathscr{R}_3: \quad A_{c''}^2 A_{x+y+z} + A_{x+y+z} B_\alpha + A_{c''} B_\beta, \quad A_{x+y+z}(A_{c''}^2 + B_\alpha + B_\beta), \tag{774b}$$
$$\mathscr{R}_4: \quad A_{c''} C_\alpha, \quad A_{x+y+z} C_\alpha, \quad A_{x+y+z} C_\gamma + A_{c''} C_{xyz}, \quad A_{x+y+z}(C_\gamma + C_{xyz}), \tag{774c}$$
$$\mathscr{R}_5: \quad A_{c''} A_{x+y+z} C_\gamma + B_\beta C_\gamma + B_\alpha C_{xyz}, \quad B_\beta(C_\alpha + C_\gamma + C_{xyz}), \tag{774d}$$
$$\mathscr{R}_6: \quad A_{c''}^3 A_{x+y+z} B_\alpha + A_{c''} A_{x+y+z} B_\alpha^2 + B_\alpha^2 B_\beta + B_\alpha B_\beta^2 + C_\alpha C_\gamma,$$
$$A_{c''}^5 A_{x+y+z} + A_{c''} A_{x+y+z} B_\alpha^2 + B_\alpha B_\beta^2 + B_\beta^3 + C_\alpha C_{xyz}, \quad A_{c''} A_{x+y+z} B_\alpha^2 + B_\alpha^2 B_\beta + A_{c''} B_\alpha C_\gamma + C_\gamma^2,$$
$$A_{c''}^3 A_{x+y+z} B_\alpha + B_\alpha B_\beta^2 + A_{x+y+z} B_\alpha C_\gamma + C_\gamma C_{xyz},$$
$$A_{c''}^5 A_{x+y+z} + A_{c''}^3 A_{x+y+z} B_\alpha + A_{c''} A_{x+y+z} B_\alpha^2 + B_\beta^3 + A_{x+y+z} B_\alpha C_\gamma + C_{xyz}^2. \tag{774e}$$

We have the following table regarding IWPs and group cohomology at degree 3.

| Wyckoff position | Little group | | Coordinates $(0,0,0)+$ $(1/2,1/2,1/2)+$ | LSM anomaly class | Topo. inv. |
|---|---|---|---|---|---|
| | Intl. | Schönflies | | | |
| 2a | 432 | $O$ | $(0,0,0)$ | $A_{c''} B_\alpha + A_{x+y+z} B_\alpha + C_\alpha + C_\gamma + C_{xyz}$ | $\varphi_2[C_2, C_2']$ |
| 6b | 422 | $D_4$ | $(0,1/2,1/2), (1/2,0,1/2),$ $(1/2,1/2,0)$ | $C_\gamma + C_{xyz}$ | $\varphi_2[C_2, T_1 T_2 C_2']$ |
| 8c | 32 | $D_3$ | $(1/4,1/4,1/4), (3/4,3/4,1/4),$ $(3/4,1/4,3/4), (1/4,3/4,3/4)$ | $C_{xyz}$ | $\varphi_2[T_1 T_2^{-1}, T_1 T_2 T_3 C_2'']$ |
| 12d | 222 | $D_2$ | $(1/4,1/2,0), (3/4,1/2,0),$ $(0,1/4,1/2), (0,3/4,1/2),$ $(1/2,0,1/4), (1/2,0,3/4)$ | $A_{x+y+z} B_\alpha$ | $\varphi_2[T_2 T_3 C_2, T_1 T_2 T_3 C_2'']$ |

**No. 212:** $P4_332$

This group is generated by three translations $T_{1,2,3}$ as given in Eqs. (88), a two-fold screw $S_2$, a two-fold screw $S_2'$, a three-fold rotation $C_3$, and a two-fold rotation $C_2''$:

$$S_2\colon (x,y,z) \to (-x+1/2, -y, z+1/2), \tag{775a}$$
$$S_2'\colon (x,y,z) \to (-x, y+1/2, -z+1/2), \tag{775b}$$
$$C_3\colon (x,y,z) \to (z,x,y), \tag{775c}$$
$$C_2''\colon (x,y,z) \to (-y+1/4, -x+1/4, -z+1/4). \tag{775d}$$

The $\mathbb{Z}_2$ cohomology ring is given by

$$\mathbb{Z}_2[A_{c''}, C_\gamma]/\langle \mathscr{R}_6 \rangle \tag{776}$$

where the relations are

$$\mathscr{R}_6\colon \quad C_\gamma^2. \tag{777a}$$

We have the following table regarding IWPs and group cohomology at degree 3.

| Wyckoff position | Little group Intl. | Schönflies | Coordinates | LSM anomaly class | Topo. inv. |
|---|---|---|---|---|---|
| 4a | 32 | $D_3$ | $(1/8,1/8,1/8)$, $(3/8,7/8,5/8)$, $(7/8,5/8,3/8)$, $(5/8,3/8,7/8)$ | $C_\gamma$ | $\varphi_2[T_1T_2^{-1}, C_2'']$ |
| 4b | 32 | $D_3$ | $(5/8,5/8,5/8)$, $(7/8,3/8,1/8)$, $(3/8,1/8,7/8)$, $(1/8,7/8,3/8)$ | Same as 4a | Same as 4a |

**No. 213:** $P4_132$

This group is generated by three translations $T_{1,2,3}$ as given in Eqs. (88), a two-fold screw $S_2$, a two-fold screw $S_2'$, a three-fold rotation $C_3$, and a two-fold rotation $C_2''$:

$$S_2\colon (x,y,z) \to (-x+1/2, -y, z+1/2), \tag{778a}$$
$$S_2'\colon (x,y,z) \to (-x, y+1/2, -z+1/2), \tag{778b}$$
$$C_3\colon (x,y,z) \to (z,x,y), \tag{778c}$$
$$C_2''\colon (x,y,z) \to (-y+3/4, -x+3/4, -z+3/4). \tag{778d}$$

The $\mathbb{Z}_2$ cohomology ring is given by

$$\mathbb{Z}_2[A_{c''}, C_\gamma]/\langle \mathscr{R}_6 \rangle \tag{779}$$

where the relations are

$$\mathscr{R}_6\colon \quad C_\gamma^2. \tag{780a}$$

We have the following table regarding IWPs and group cohomology at degree 3.

| Wyckoff position | Little group Intl. | Schönflies | Coordinates | LSM anomaly class | Topo. inv. |
|---|---|---|---|---|---|
| 4a | 32 | $D_3$ | $(3/8,3/8,3/8)$, $(1/8,5/8,7/8)$, $(5/8,7/8,1/8)$, $(7/8,1/8,5/8)$ | $C_\gamma$ | $\varphi_2[T_1T_2^{-1}, C_2'']$ |
| 4b | 32 | $D_3$ | $(7/8,7/8,7/8)$, $(5/8,1/8,3/8)$, $(1/8,3/8,5/8)$, $(3/8,5/8,1/8)$ | Same as 4a | Same as 4a |

**No. 214:** $I4_132$

This group is generated by three translations $T_{1,2,3}$ as given in Eqs. (91), a two-fold rotation $C_2$, a two-fold rotation $C_2'$, a three-fold rotation $C_3$, and a two-fold rotation $C_2''$:

$$C_2 : (x, y, z) \rightarrow (-x, -y + 1/2, z), \tag{781a}$$
$$C_2' : (x, y, z) \rightarrow (-x + 1/2, y, -z), \tag{781b}$$
$$C_3 : (x, y, z) \rightarrow (z, x, y), \tag{781c}$$
$$C_2'' : (x, y, z) \rightarrow (-y + 1/4, -x + 1/4, -z + 1/4). \tag{781d}$$

The $\mathbb{Z}_2$ cohomology ring is given by

$$\mathbb{Z}_2[A_{c''}, A_{x+y+z}, C_\gamma]/\langle \mathcal{R}_6 \rangle \tag{782}$$

where the relations are

$$\mathcal{R}_6 : \quad C_\gamma(A_{c''}^2 A_{x+y+z} + A_{c''} A_{x+y+z}^2 + C_\gamma). \tag{783a}$$

We have the following table regarding IWPs and group cohomology at degree 3.

| Wyckoff position | Little group Intl. | Little group Schönflies | Coordinates $(0,0,0)+$ $(1/2,1/2,1/2)+$ | LSM anomaly class | Topo. inv. |
|---|---|---|---|---|---|
| 8a | 32 | $D_3$ | $(1/8, 1/8, 1/8)$, $(3/8, 7/8, 5/8)$, $(7/8, 5/8, 3/8)$, $(5/8, 3/8, 7/8)$ | N/A | N/A |
| 8b | 32 | $D_3$ | $(7/8, 7/8, 7/8)$, $(5/8, 1/8, 3/8)$, $(1/8, 3/8, 5/8)$, $(3/8, 5/8, 1/8)$ | N/A | N/A |
| 12c | 222 | $D_2$ | $(1/8, 0, 1/4)$, $(3/8, 0, 3/4)$, $(1/4, 1/8, 0)$, $(3/4, 3/8, 0)$, $(0, 1/4, 1/8)$, $(0, 3/4, 3/8)$ | $A_{c''}^2 A_{x+y+z} + A_{c''} A_{x+y+z}^2 + C_\gamma$ | $\varphi_2[C_2, C_2'']$ |
| 12d | 222 | $D_2$ | $(5/8, 0, 1/4)$, $(7/8, 0, 3/4)$, $(1/4, 5/8, 0)$, $(3/4, 7/8, 0)$, $(0, 1/4, 5/8)$, $(0, 3/4, 7/8)$ | $C_\gamma$ | $\varphi_2[C_2, T_1 T_2 C_2'']$ |

**No. 215:** $P\bar{4}3m$

This group is generated by three translations $T_{1,2,3}$ as given in Eqs. (88), a two-fold rotation $C_2$, a two-fold rotation $C_2'$, a three-fold rotation $C_3$, and a mirror $M$:

$$C_2 : (x, y, z) \rightarrow (-x, -y, z), \tag{784a}$$
$$C_2' : (x, y, z) \rightarrow (-x, y, -z), \tag{784b}$$
$$C_3 : (x, y, z) \rightarrow (z, x, y), \tag{784c}$$
$$M : (x, y, z) \rightarrow (y, x, z). \tag{784d}$$

The $\mathbb{Z}_2$ cohomology ring is given by

$$\mathbb{Z}_2[A_m, A_{x+y+z}, B_\alpha, B_{xy+xz+yz}, C_\alpha, C_{xyz}]/\langle \mathcal{R}_3, \mathcal{R}_4, \mathcal{R}_5, \mathcal{R}_6 \rangle \tag{785}$$

where the relations are

$$\mathcal{R}_3 : \quad A_m A_{x+y+z}^2, \tag{786a}$$
$$\mathcal{R}_4 : \quad A_m C_\alpha, \quad A_{x+y+z}(A_{x+y+z}^3 + A_{x+y+z} B_\alpha + C_\alpha), \quad A_m(A_{x+y+z} B_{xy+xz+yz} + C_{xyz}), \quad A_{x+y+z} C_{xyz},$$
$$\qquad B_{xy+xz+yz}(A_{x+y+z}^2 + B_\alpha + B_{xy+xz+yz}), \tag{786b}$$
$$\mathcal{R}_5 : \quad A_{x+y+z}^3 B_{xy+xz+yz} + A_{x+y+z} B_\alpha B_{xy+xz+yz} + B_{xy+xz+yz} C_\alpha + B_\alpha C_{xyz},$$
$$\qquad B_{xy+xz+yz}(A_{x+y+z}^3 + A_{x+y+z} B_\alpha + C_\alpha + C_{xyz}), \tag{786c}$$
$$\mathcal{R}_6 : \quad C_{xyz}(C_\alpha + C_{xyz}). \tag{786d}$$

We have the following table regarding IWPs and group cohomology at degree 3.

| Wyckoff position | Little group Intl. | Little group Schönflies | Coordinates | LSM anomaly class | Topo. inv. |
|---|---|---|---|---|---|
| 1a | $\bar{4}3m$ | $T_d$ | $(0,0,0)$ | $A_{x+y+z}^3 + A_{x+y+z}B_\alpha + C_\alpha + C_{xyz}$ | $\varphi_2[C_2, C_2']$ |
| 1b | $\bar{4}3m$ | $T_d$ | $(1/2,1/2,1/2)$ | $C_{xyz}$ | $\varphi_2[T_1T_2C_2, T_1T_3C_2']$ |
| 3c | $\bar{4}2m$ | $D_{2d}$ | $(0,1/2,1/2), (1/2,0,1/2),$ $(1/2,1/2,0)$ | $A_{x+y+z}B_{xy+xz+yz}$ | $\varphi_2[T_1T_2C_2, T_1C_2']$ |
| 3d | $\bar{4}2m$ | $D_{2d}$ | $(1/2,0,0), (0,1/2,0),$ $(0,0,1/2)$ | $A_{x+y+z}(A_{x+y+z}^2 + B_\alpha + B_{xy+xz+yz})$ | $\varphi_2[T_1C_2, T_1C_2']$ |

## No. 216: $F\bar{4}3m$

This group is generated by three translations $T_{1,2,3}$ as given in Eqs. (92), a two-fold rotation $C_2$, a two-fold rotation $C_2'$, a three-fold rotation $C_3$, and a mirror $M$:

$$C_2: (x,y,z) \rightarrow (-x,-y,z), \tag{787a}$$
$$C_2': (x,y,z) \rightarrow (-x,y,-z), \tag{787b}$$
$$C_3: (x,y,z) \rightarrow (z,x,y), \tag{787c}$$
$$M: (x,y,z) \rightarrow (y,x,z). \tag{787d}$$

The $\mathbb{Z}_2$ cohomology ring is given by

$$\mathbb{Z}_2[A_m, B_\alpha, B_\beta, B_{xy+xz+yz}, C_\alpha, C_\beta, C_\gamma, C_{xyz}]/\langle \mathscr{R}_4, \mathscr{R}_5, \mathscr{R}_6 \rangle \tag{788}$$

where the relations are

$$\mathscr{R}_4: \quad A_mC_\alpha, \quad A_mC_\beta, \quad B_\alpha B_{xy+xz+yz} + A_mC_\gamma, \quad B_\alpha B_\beta + B_\beta^2 + A_mC_\gamma, \quad A_m^2 B_\beta + B_\beta B_{xy+xz+yz} + A_m C_{xyz}, \quad B_{xy+xz+yz}^2, \tag{789a}$$

$$\mathscr{R}_5: \quad B_{xy+xz+yz}C_\alpha, \quad B_\beta C_\alpha + B_\alpha C_\beta, \quad B_\beta(C_\alpha + C_\beta), \quad B_{xy+xz+yz}C_\beta, \quad B_{xy+xz+yz}C_\gamma, \quad A_m B_\alpha B_\beta + B_\beta C_\gamma + B_\alpha C_{xyz},$$
$$A_m B_\alpha B_\beta + A_m^2 C_\gamma + B_\beta C_\gamma + B_\beta C_{xyz}, \quad A_m^3 B_\beta + A_m^2 C_{xyz} + B_{xy+xz+yz}C_{xyz}, \tag{789b}$$

$$\mathscr{R}_6: \quad C_\beta(C_\alpha + C_\beta), \quad C_\beta C_\gamma + C_\alpha C_{xyz}, \quad (C_\alpha + C_\beta)C_{xyz}, \quad C_\gamma(C_\alpha + C_\gamma), \quad A_m B_\beta C_\gamma + C_\alpha C_{xyz} + C_\gamma C_{xyz},$$
$$A_m^2 B_\alpha B_\beta + A_m^3 C_\gamma + C_\alpha C_{xyz} + C_{xyz}^2. \tag{789c}$$

We have the following table regarding IWPs and group cohomology at degree 3.

| Wyckoff position | Little group Intl. | Little group Schönflies | Coordinates $(0,0,0)+ (0,1/2,1/2)+$ $(1/2,0,1/2)+ (1/2,1/2,0)+$ | LSM anomaly class | Topo. inv. |
|---|---|---|---|---|---|
| 4a | $\bar{4}3m$ | $T_d$ | $(0,0,0)$ | $C_\alpha + C_\gamma + C_{xyz}$ | $\varphi_2[C_2, C_2']$ |
| 4b | $\bar{4}3m$ | $T_d$ | $(1/2,1/2,1/2)$ | $C_\beta + C_\gamma + C_{xyz}$ | $\varphi_2[C_2, T_1T_2T_3^{-1}C_2']$ |
| 4c | $\bar{4}3m$ | $T_d$ | $(1/4,1/4,1/4)$ | $C_\beta + C_{xyz}$ | $\varphi_2[T_3C_2, T_2C_2']$ |
| 4d | $\bar{4}3m$ | $T_d$ | $(3/4,3/4,3/4)$ | $C_{xyz}$ | $\varphi_2[T_3^{-1}C_2, T_2^{-1}C_2']$ |

## No. 217: $I\bar{4}3m$

This group is generated by three translations $T_{1,2,3}$ as given in Eqs. (91), a two-fold rotation $C_2$, a two-fold rotation $C_2'$, a three-fold rotation $C_3$, and a mirror $M$:

$$C_2: (x,y,z) \rightarrow (-x,-y,z), \tag{790a}$$
$$C_2': (x,y,z) \rightarrow (-x,y,-z), \tag{790b}$$
$$C_3: (x,y,z) \rightarrow (z,x,y), \tag{790c}$$
$$M: (x,y,z) \rightarrow (y,x,z). \tag{790d}$$

The $\mathbb{Z}_2$ cohomology ring is given by

$$\mathbb{Z}_2[A_m, A_{x+y+z}, B_\alpha, B_\beta, C_\alpha, C_{xyz}]/\langle \mathscr{R}_2, \mathscr{R}_3, \mathscr{R}_4, \mathscr{R}_5, \mathscr{R}_6 \rangle \tag{791}$$

where the relations are

$$\mathcal{R}_2: \quad A_{x+y+z}^2, \tag{792a}$$

$$\mathcal{R}_3: \quad A_{x+y+z}B_\alpha + A_m B_\beta, \quad A_{x+y+z}(B_\alpha + B_\beta), \tag{792b}$$

$$\mathcal{R}_4: \quad A_m C_\alpha, \quad A_{x+y+z}C_\alpha, \quad A_{x+y+z}C_{xyz}, \tag{792c}$$

$$\mathcal{R}_5: \quad A_{x+y+z}B_\alpha^2 + B_\beta C_\alpha + B_\beta C_{xyz}, \tag{792d}$$

$$\mathcal{R}_6: \quad B_\alpha^2 B_\beta + B_\beta^3 + C_\alpha C_{xyz}, \quad B_\alpha^2 B_\beta + B_\beta^3 + C_{xyz}^2. \tag{792e}$$

We have the following table regarding IWPs and group cohomology at degree 3.

| Wyckoff position | Little group Intl. | Schönflies | Coordinates $(0,0,0)+$ $(1/2,1/2,1/2)+$ | LSM anomaly class | Topo. inv. |
|---|---|---|---|---|---|
| 2a | $\bar{4}3m$ | $T_d$ | $(0,0,0)$ | $A_{x+y+z}B_\alpha + C_\alpha + C_{xyz}$ | $\varphi_2[C_2, C_2']$ |
| 6b | $\bar{4}2m$ | $D_{2d}$ | $(0,1/2,1/2)$, $(1/2,0,1/2)$, $(1/2,1/2,0)$ | $C_{xyz}$ | $\varphi_2[C_2, T_1 T_2 C_2']$ |
| 12d | $\bar{4}$ | $S_4$ | $(1/4,1/2,0)$, $(3/4,1/2,0)$, $(0,1/4,1/2)$, $(0,3/4,1/2)$, $(1/2,0,1/4)$, $(1/2,0,3/4)$ | $A_{x+y+z}B_\alpha$ | $\varphi_2[T_1 T_2 T_3 C_2' M, T_1 T_3 C_2]$ |

## No. 218: $P\bar{4}3n$

This group is generated by three translations $T_{1,2,3}$ as given in Eqs. (88), a two-fold rotation $C_2$, a two-fold rotation $C_2'$, a three-fold rotation $C_3$, and a glide $G$:

$$C_2: (x,y,z) \to (-x,-y,z), \tag{793a}$$

$$C_2': (x,y,z) \to (-x,y,-z), \tag{793b}$$

$$C_3: (x,y,z) \to (z,x,y), \tag{793c}$$

$$G: (x,y,z) \to (y+1/2, x+1/2, z+1/2). \tag{793d}$$

The $\mathbb{Z}_2$ cohomology ring is given by

$$\mathbb{Z}_2[A_m, B_\alpha, B_\beta, B_{xy+xz+yz}, C_\alpha, C_\beta, C_\gamma]/\langle \mathcal{R}_2, \mathcal{R}_3, \mathcal{R}_4, \mathcal{R}_5, \mathcal{R}_6\rangle \tag{794}$$

where the relations are

$$\mathcal{R}_2: \quad A_m^2, \tag{795a}$$

$$\mathcal{R}_3: \quad A_m B_\beta, \tag{795b}$$

$$\mathcal{R}_4: \quad A_m C_\alpha, \quad A_m C_\beta, \quad A_m C_\gamma, \quad B_\beta^2 + B_\alpha B_{xy+xz+yz} + B_\beta B_{xy+xz+yz} + B_{xy+xz+yz}^2, \tag{795c}$$

$$\mathcal{R}_5: \quad A_m B_\alpha^2 + B_\alpha C_\alpha + B_{xy+xz+yz}C_\alpha + B_\alpha C_\beta + B_\beta C_\beta,$$
$$A_m B_\alpha B_{xy+xz+yz} + B_\beta C_\alpha + B_{xy+xz+yz}C_\alpha + B_\alpha C_\beta + B_{xy+xz+yz}C_\beta + B_\alpha C_\gamma,$$
$$A_m B_\alpha^2 + A_m B_\alpha B_{xy+xz+yz} + B_\alpha C_\alpha + B_{xy+xz+yz}C_\alpha + B_\alpha C_\beta + B_{xy+xz+yz}C_\beta + B_\beta C_\gamma,$$
$$A_m B_\alpha^2 + B_\alpha C_\alpha + B_\beta C_\alpha + B_\alpha C_\beta + B_{xy+xz+yz}C_\beta + B_{xy+xz+yz}C_\gamma, \tag{795d}$$

$$\mathcal{R}_6: \quad B_\alpha B_\beta^2 + B_\beta^3 + C_\alpha C_\gamma, \quad B_\alpha B_\beta^2 + C_\alpha^2 + C_\alpha C_\beta + C_\beta^2, \quad B_\alpha B_\beta^2 + B_\beta^2 B_{xy+xz+yz} + C_\beta C_\gamma, \quad B_\alpha B_\beta^2 + B_\beta^3 + C_\gamma^2. \tag{795e}$$

We have the following table regarding IWPs and group cohomology at degree 3.

| Wyckoff position | Little group Intl. | Schönflies | Coordinates | LSM anomaly class | Topo. inv. |
|---|---|---|---|---|---|
| 2a | 23 | $T$ | $(0,0,0)$, $(1/2,1/2,1/2)$ | $A_m B_\alpha + C_\alpha + C_\gamma$ | $\varphi_2[C_2, C_2']$ |
| 6b | 222 | $D_2$ | $(0,1/2,1/2)$, $(1/2,0,1/2)$, $(1/2,1/2,0)$, $(0,1/2,0)$, $(1/2,0,0)$, $(0,0,1/2)$ | $C_\gamma$ | $\varphi_2[C_2, T_3 C_2']$ |
| 6c | $\bar{4}$ | $S_4$ | $(1/4,1/2,0)$, $(3/4,1/2,0)$, $(0,1/4,1/2)$, $(0,3/4,1/2)$, $(1/2,0,1/4)$, $(1/2,0,3/4)$ | $A_m(B_\alpha + B_{xy+xz+yz})$ | $\varphi_2[C_2'G, T_1^{-1}C_2]$ |
| 6d | $\bar{4}$ | $S_4$ | $(1/4,0,1/2)$, $(3/4,0,1/2)$, $(1/2,1/4,0)$, $(1/2,3/4,0)$, $(0,1/2,1/4)$, $(0,1/2,3/4)$ | $A_m B_{xy+xz+yz}$ | $\varphi_2[T_1 C_2'G, T_2 C_2]$ |

**No. 219:** $F\bar{4}3c$

This group is generated by three translations $T_{1,2,3}$ as given in Eqs. (92), a two-fold rotation $C_2$, a two-fold rotation $C_2'$, a three-fold rotation $C_3$, and a glide $G$:

$$C_2 \colon (x, y, z) \to (-x, -y, z), \tag{796a}$$
$$C_2' \colon (x, y, z) \to (-x, y, -z), \tag{796b}$$
$$C_3 \colon (x, y, z) \to (z, x, y), \tag{796c}$$
$$G \colon (x, y, z) \to (y + 1/2, x + 1/2, z + 1/2). \tag{796d}$$

The $\mathbb{Z}_2$ cohomology ring is given by

$$\mathbb{Z}_2[A_m, B_\alpha, B_{xy+xz+yz}, C_\alpha, C_\beta, F_\gamma, F_\delta]/\langle \mathcal{R}_3, \mathcal{R}_4, \mathcal{R}_5, \mathcal{R}_6, \mathcal{R}_7, \mathcal{R}_8, \mathcal{R}_9, \mathcal{R}_{12} \rangle \tag{797}$$

where the relations are

$$\mathcal{R}_3 \colon \quad A_m^3, \tag{798a}$$
$$\mathcal{R}_4 \colon \quad A_m^2 B_\alpha, \quad A_m^2 B_{xy+xz+yz}, \quad A_m C_\alpha, \quad A_m C_\beta, \quad B_{xy+xz+yz}(B_\alpha + B_{xy+xz+yz}), \tag{798b}$$
$$\mathcal{R}_5 \colon \quad B_{xy+xz+yz}C_\alpha + B_\alpha C_\beta, \quad B_{xy+xz+yz}(C_\alpha + C_\beta), \tag{798c}$$
$$\mathcal{R}_6 \colon \quad C_\beta(C_\alpha + C_\beta), \tag{798d}$$
$$\mathcal{R}_7 \colon \quad A_m F_\gamma, \quad A_m F_\delta, \tag{798e}$$
$$\mathcal{R}_8 \colon \quad B_{xy+xz+yz}F_\gamma + B_\alpha F_\delta + B_\alpha C_\alpha C_\beta, \quad (B_{xy+xz+yz} + B_\alpha)F_\delta, \tag{798f}$$
$$\mathcal{R}_9 \colon \quad C_\beta F_\gamma + C_\beta F_\delta + C_\alpha^2 C_\beta, \quad (C_\alpha + C_\beta)F_\delta, \tag{798g}$$
$$\mathcal{R}_{12} \colon \quad F_\gamma^2 + C_\alpha^2(F_\gamma + B_\alpha^3 + C_\alpha^2), \quad F_\delta^2 + C_\alpha C_\beta(F_\delta + B_{xy+xz+yz}^3 + C_\alpha C_\beta), \quad F_\gamma F_\delta + C_\alpha C_\beta(B_\alpha^3 + C_\alpha^2). \tag{798h}$$

We have the following table regarding IWPs and group cohomology at degree 3.

| Wyckoff position | Little group Intl. | Little group Schönflies | Coordinates $(0,0,0)+$ $(0,1/2,1/2)+$ $(1/2,0,1/2)+$ $(1/2,1/2,0)+$ | LSM anomaly class | Topo. inv. |
|---|---|---|---|---|---|
| 8a | 23 | $T$ | $(0,0,0), (1/2,1/2,1/2)$ | $A_m B_\alpha + A_m B_{xy+xz+yz} + C_\alpha + C_\beta$ | $\varphi_2[C_2, C_2']$ |
| 8b | 23 | $T$ | $(1/4,1/4,1/4), (3/4,3/4,3/4)$ | $A_m B_{xy+xz+yz} + C_\beta$ | $\varphi_2[T_3 C_2, T_2 C_2']$ |
| 24c | $\bar{4}$ | $S_4$ | $(0,1/4,1/4), (0,3/4,1/4),$ $(1/4,0,1/4), (1/4,0,3/4),$ $(1/4,1/4,0), (3/4,1/4,0)$ | $A_m B_{xy+xz+yz}$ | $\varphi_2[T_2 C_2' G, T_1 T_2^{-1} C_2]$ |
| 24d | $\bar{4}$ | $S_4$ | $(1/4,0,0), (3/4,0,0),$ $(0,1/4,0), (0,3/4,0),$ $(0,0,1/4), (0,0,3/4)$ | $A_m(B_\alpha + B_{xy+xz+yz})$ | $\varphi_2[T_2^2 T_3^{-1} C_2' G, C_2]$ |

**No. 220:** $I\bar{4}3d$

This group is generated by three translations $T_{1,2,3}$ as given in Eqs. (91), a two-fold rotation $C_2$, a two-fold rotation $C_2'$, a three-fold rotation $C_3$, and a glide $G$:

$$C_2 \colon (x, y, z) \to (-x, -y + 1/2, z), \tag{799a}$$
$$C_2' \colon (x, y, z) \to (-x + 1/2, y, -z), \tag{799b}$$
$$C_3 \colon (x, y, z) \to (z, x, y), \tag{799c}$$
$$G \colon (x, y, z) \to (y + 1/4, x + 1/4, z + 1/4). \tag{799d}$$

The $\mathbb{Z}_2$ cohomology ring is given by

$$\mathbb{Z}_2[A_m, B_\alpha, C_\gamma]/\langle \mathcal{R}_2, \mathcal{R}_6 \rangle \tag{800}$$

where the relations are

$$\mathcal{R}_2 \colon \quad A_m^2, \tag{801a}$$
$$\mathcal{R}_6 \colon \quad C_\gamma(A_m B_\alpha + C_\gamma). \tag{801b}$$

We have the following table regarding IWPs and group cohomology at degree 3.

| Wyckoff position | Little group Intl. | Little group Schönflies | Coordinates $(0,0,0) + (1/2,1/2,1/2)+$ | LSM anomaly class | Topo. inv. |
|---|---|---|---|---|---|
| 12a | $\bar{4}$ | $S_4$ | $(3/8,0,1/4),\ (1/8,0,3/4),$ $(1/4,3/8,0),\ (3/4,1/8,0),$ $(0,1/4,3/8),\ (0,3/4,1/8)$ | $C_\gamma$ | $\varphi_2[T_1 T_2 C_2' G, C_2]$ |
| 12b | $\bar{4}$ | $S_4$ | $(7/8,0,1/4),\ (5/8,0,3/4),$ $(1/4,7/8,0),\ (3/4,5/8,0),$ $(0,1/4,7/8),\ (0,3/4,5/8)$ | $A_m B_\alpha + C_\gamma$ | $\varphi_2[C_2' G, C_2]$ |
| 16c | $3$ | $C_3$ | $(x,x,x),\ (-x+1/2,-x,x+1/2),$ $(-x,x+1/2,-x+1/2),\ (x+1/2,-x+1/2,-x),$ $(x+1/4,x+1/4,x+1/4),\ (-x+1/4,-x+3/4,x+3/4),$ $(x+3/4,-x+1/4,-x+3/4),\ (-x+3/4,x+3/4,-x+1/4)$ | N/A | N/A |

**No. 221:** $Pm\bar{3}m$

This group is generated by three translations $T_{1,2,3}$ as given in Eqs. (88), a two-fold rotation $C_2$, a two-fold rotation $C_2'$, a three-fold rotation $C_3$, a mirror $M$, and an inversion $I$:

$$C_2 : (x,y,z) \to (-x,-y,z), \tag{802a}$$
$$C_2' : (x,y,z) \to (-x,y,-z), \tag{802b}$$
$$C_3 : (x,y,z) \to (z,x,y), \tag{802c}$$
$$M : (x,y,z) \to (y,x,z), \tag{802d}$$
$$I : (x,y,z) \to (-x,-y,-z). \tag{802e}$$

The $\mathbb{Z}_2$ cohomology ring is given by

$$\mathbb{Z}_2[A_i, A_m, A_{x+y+z}, B_\alpha, B_{xy+xz+yz}, C_\alpha, C_{xyz}]/\langle \mathscr{R}_3, \mathscr{R}_4, \mathscr{R}_5, \mathscr{R}_6 \rangle \tag{803}$$

where the relations are

$$\mathscr{R}_3: \quad A_m A_{x+y+z}(A_i + A_{x+y+z}), \tag{804a}$$

$$\mathscr{R}_4: \quad A_i^2 A_{x+y+z}^2 + A_i A_{x+y+z}^3 + A_{x+y+z}^4 + A_i A_{x+y+z} B_\alpha + A_{x+y+z}^2 B_\alpha + A_i A_{x+y+z} B_{xy+xz+yz} + A_{x+y+z} C_\alpha + A_i C_{xyz},$$
$$A_m C_\alpha, \quad A_m(A_i^2 A_{x+y+z} + A_{x+y+z} B_{xy+xz+yz} + C_{xyz}),$$
$$A_{x+y+z}(A_i^3 + A_i A_{x+y+z}^2 + A_{x+y+z}^3 + A_i B_\alpha + A_{x+y+z} B_\alpha + A_i B_{xy+xz+yz} + C_\alpha + C_{xyz}),$$
$$B_{xy+xz+yz}(A_i^2 + A_i A_m + A_i A_{x+y+z} + A_{x+y+z}^2 + B_\alpha + B_{xy+xz+yz}), \tag{804b}$$

$$\mathscr{R}_5: \quad A_i^4 A_{x+y+z} + A_i^3 A_{x+y+z}^2 + A_i^2 A_{x+y+z}^3 + A_i A_{x+y+z}^4 + A_i A_{x+y+z}^2 B_\alpha + A_i^2 A_{x+y+z} B_{xy+xz+yz}$$
$$+ A_{x+y+z}^3 B_{xy+xz+yz} + A_{x+y+z} B_\alpha B_{xy+xz+yz} + A_i A_{x+y+z} C_\alpha + B_{xy+xz+yz} C_\alpha + B_\alpha C_{xyz},$$
$$B_{xy+xz+yz}(A_i^2 A_{x+y+z} + A_i A_m A_{x+y+z} + A_{x+y+z}^3 + A_{x+y+z} B_\alpha + C_\alpha + C_{xyz}), \tag{804c}$$

$$\mathscr{R}_6: \quad A_i^5 A_{x+y+z} + A_i^3 A_{x+y+z}^3 + A_i^2 A_{x+y+z}^4 + A_i A_{x+y+z}^3 B_\alpha + A_{x+y+z}^4 B_\alpha + A_i A_{x+y+z} B_\alpha^2 + A_{x+y+z}^2 B_\alpha^2$$
$$+ A_i^3 A_{x+y+z} B_{xy+xz+yz} + A_i^2 A_m A_{x+y+z} B_{xy+xz+yz} + A_i A_{x+y+z} B_\alpha B_{xy+xz+yz} + A_{x+y+z} B_\alpha C_\alpha + C_\alpha C_{xyz} + C_{xyz}^2. \tag{804d}$$

We have the following table regarding IWPs and group cohomology at degree 3.

| Wyckoff position | Little group Intl. | Little group Schönflies | Coordinates | LSM anomaly class | Topo. inv. |
|---|---|---|---|---|---|
| 1a | $m\bar{3}m$ | $O_h$ | $(0,0,0)$ | $A_i^3 + A_i^2 A_m + A_i A_m A_{x+y+z} + A_i A_{x+y+z}^2 + A_{x+y+z}^3$ $+ A_i B_\alpha + A_{x+y+z} B_\alpha + A_i B_{xy+xz+yz} + C_\alpha + C_{xyz}$ | $\varphi_1[I]$ |
| 1b | $m\bar{3}m$ | $O_h$ | $(1/2,1/2,1/2)$ | $A_i^2 A_{x+y+z} + C_{xyz}$ | $\varphi_1[T_1 T_2 T_3 I]$ |
| 3c | $4/mmm$ | $D_{4h}$ | $(0,1/2,1/2),\ (1/2,0,1/2),$ $(1/2,1/2,0)$ | $(A_i + A_{x+y+z}) B_{xy+xz+yz}$ | $\varphi_2[T_1 T_2 C_2, T_1 C_2']$ |
| 3d | $4/mmm$ | $D_{4h}$ | $(1/2,0,0),\ (0,1/2,0),$ $(0,0,1/2)$ | $A_{x+y+z}(A_i^2 + A_i A_m + A_i A_{x+y+z} + A_{x+y+z}^2 + B_\alpha + B_{xy+xz+yz})$ | $\varphi_2[C_2, T_3 C_2']$ |

**No. 222: $Pn\bar{3}n$**

This group is generated by three translations $T_{1,2,3}$ as given in Eqs. (88), a two-fold rotation $C_2$, a two-fold rotation $C_2'$, a three-fold rotation $C_3$, a glide $G$, and an inversion $I$:

$$C_2\colon (x,y,z) \to (-x+1/2, -y+1/2, z), \tag{805a}$$
$$C_2'\colon (x,y,z) \to (-x+1/2, y, -z+1/2), \tag{805b}$$
$$C_3\colon (x,y,z) \to (z,x,y), \tag{805c}$$
$$G\colon (x,y,z) \to (y+1/2, x+1/2, z+1/2), \tag{805d}$$
$$I\colon (x,y,z) \to (-x,-y,-z). \tag{805e}$$

The $\mathbb{Z}_2$ cohomology ring is given by

$$\mathbb{Z}_2[A_i, A_m, B_\alpha, B_\beta, C_\alpha, C_\gamma]/\langle \mathcal{R}_2, \mathcal{R}_3, \mathcal{R}_4, \mathcal{R}_5, \mathcal{R}_6 \rangle \tag{806}$$

where the relations are

$$\mathcal{R}_2\colon \quad A_m(A_i + A_m), \tag{807a}$$
$$\mathcal{R}_3\colon \quad A_i(B_\alpha + B_\beta), \quad A_iB_\alpha + A_mB_\beta, \tag{807b}$$
$$\mathcal{R}_4\colon \quad A_iC_\alpha, \quad A_mC_\alpha, \quad (A_i + A_m)C_\gamma, \tag{807c}$$
$$\mathcal{R}_5\colon \quad (B_\alpha + B_\beta)(C_\alpha + C_\gamma), \tag{807d}$$
$$\mathcal{R}_6\colon \quad B_\alpha B_\beta^2 + B_\beta^3 + C_\alpha C_\gamma, \quad A_i^5 A_m + A_i^4 B_\alpha + B_\alpha B_\beta^2 + B_\beta^3 + A_i B_\alpha C_\gamma + C_\gamma^2. \tag{807e}$$

We have the following table regarding IWPs and group cohomology at degree 3.

| Wyckoff position | Little group Intl. | Little group Schönflies | Coordinates | LSM anomaly class | Topo. inv. |
|---|---|---|---|---|---|
| 2a | 432 | $O$ | $(1/4,1/4,1/4)$, $(3/4,3/4,3/4)$ | $A_i^2 A_m + A_i B_\alpha + C_\alpha + C_\gamma$ | $\varphi_2[C_2, C_2']$ |
| 6b | 422 | $D_4$ | $(3/4,1/4,1/4)$, $(1/4,3/4,1/4)$, $(1/4,1/4,3/4)$, $(1/4,3/4,3/4)$, $(3/4,1/4,3/4)$, $(3/4,3/4,1/4)$ | $A_i^2 A_m + A_i B_\alpha + A_m B_\alpha + C_\gamma$ | $\varphi_2[C_2, T_3 C_2']$ |
| 8c | $\bar{3}$ | $C_{3i}$ | $(0,0,0)$, $(1/2,1/2,0)$, $(1/2,0,1/2)$, $(0,1/2,1/2)$, $(0,0,1/2)$, $(1/2,1/2,1/2)$, $(0,1/2,0)$, $(1/2,0,0)$ | $A_i^2(A_i + A_m)$ | $\varphi_1[I]$ |
| 12d | $\bar{4}$ | $S_4$ | $(0,3/4,1/4)$, $(1/2,3/4,1/4)$, $(1/4,0,3/4)$, $(1/4,1/2,3/4)$, $(3/4,1/4,0)$, $(3/4,1/4,1/2)$, $(3/4,0,1/4)$, $(3/4,1/2,1/4)$, $(0,1/4,3/4)$, $(1/2,1/4,3/4)$, $(1/4,3/4,1/2)$, $(1/4,3/4,0)$ | $(A_i + A_m)B_\alpha$ | $\varphi_2[C_2'G, T_1^{-1}C_2]$ |

**No. 223: $Pm\bar{3}n$**

This group is generated by three translations $T_{1,2,3}$ as given in Eqs. (88), a two-fold rotation $C_2$, a two-fold rotation $C_2'$, a three-fold rotation $C_3$, a glide $G$, and an inversion $I$:

$$C_2\colon (x,y,z) \to (-x,-y,z), \tag{808a}$$
$$C_2'\colon (x,y,z) \to (-x,y,-z), \tag{808b}$$
$$C_3\colon (x,y,z) \to (z,x,y), \tag{808c}$$
$$G\colon (x,y,z) \to (y+1/2, x+1/2, z+1/2), \tag{808d}$$
$$I\colon (x,y,z) \to (-x,-y,-z). \tag{808e}$$

The $\mathbb{Z}_2$ cohomology ring is given by

$$\mathbb{Z}_2[A_i, A_m, B_\alpha, B_\beta, B_{xy+xz+yz}, C_\alpha, C_\beta, C_\gamma]/\langle \mathcal{R}_2, \mathcal{R}_3, \mathcal{R}_4, \mathcal{R}_5, \mathcal{R}_6 \rangle \tag{809}$$

where the relations are

$$\mathcal{R}_2: \quad A_m(A_i + A_m), \tag{810a}$$

$$\mathcal{R}_3: \quad A_m(A_i^2 + B_\beta), \tag{810b}$$

$$\mathcal{R}_4: \quad A_m C_\alpha, \quad A_m(A_i B_\alpha + C_\beta), \quad A_m(A_i B_{xy+xz+yz} + C_\gamma),$$
$$A_i^3 A_m + A_i^2 B_\alpha + A_i A_m B_\alpha + B_\beta^2 + A_i^2 B_{xy+xz+yz} + A_i A_m B_{xy+xz+yz}$$
$$+ B_\alpha B_{xy+xz+yz} + B_\beta B_{xy+xz+yz} + B_{xy+xz+yz}^2 + A_i C_\gamma, \tag{810c}$$

$$\mathcal{R}_5: \quad A_i^5 + A_i^3 B_\alpha + A_m B_\alpha^2 + A_i B_\alpha B_\beta + A_i B_\beta^2 + A_i^2 C_\alpha + B_\alpha C_\alpha + B_{xy+xz+yz} C_\alpha + A_i^2 C_\beta + B_\alpha C_\beta + B_\beta C_\beta,$$
$$A_i^3 B_\alpha + A_i^2 A_m B_\alpha + A_m B_\alpha^2 + A_i^3 B_\beta + A_i B_\alpha B_\beta + A_i B_\beta^2 + A_i^3 B_{xy+xz+yz} + A_i B_\beta B_{xy+xz+yz}$$
$$+ B_\beta C_\alpha + B_{xy+xz+yz} C_\alpha + A_i^2 C_\beta + B_\alpha C_\beta + B_{xy+xz+yz} C_\beta + B_\alpha C_\gamma,$$
$$A_i^5 + A_m B_\alpha^2 + A_i B_\alpha B_\beta + A_i B_\beta^2 + A_i^3 B_{xy+xz+yz} + A_m B_\alpha B_{xy+xz+yz} + A_i B_\beta B_{xy+xz+yz} + A_i^2 C_\alpha$$
$$+ B_\alpha C_\alpha + B_{xy+xz+yz} C_\alpha + A_i^2 C_\beta + B_\alpha C_\beta + B_{xy+xz+yz} C_\beta + A_i^2 C_\gamma + B_\beta C_\gamma,$$
$$A_i^5 + A_i^4 A_m + A_m B_\alpha^2 + A_i^3 B_\beta + A_i B_\alpha B_\beta + A_i B_\beta^2 + A_i B_\beta B_{xy+xz+yz} + A_i^2 C_\alpha + B_\alpha C_\alpha$$
$$+ B_\beta C_\beta + A_i^2 C_\beta + B_\alpha C_\beta + B_{xy+xz+yz} C_\beta + A_i^2 C_\gamma + B_{xy+xz+yz} C_\gamma, \tag{810d}$$

$$\mathcal{R}_6: \quad A_i^6 + A_i^2 B_\alpha^2 + A_i A_m B_\alpha^2 + A_i^4 B_\beta + A_i^2 B_\alpha B_\beta + A_i^2 B_\beta^2 + B_\alpha B_\beta^2 + B_\beta^3 + A_i^3 C_\alpha + A_i B_\beta C_\alpha + C_\alpha C_\gamma,$$
$$A_i^6 + A_i^4 B_\alpha + A_i^2 B_\alpha^2 + A_i^2 B_\beta^2 + B_\alpha B_\beta^2 + C_\alpha^2 + C_\alpha C_\beta + C_\beta^2,$$
$$A_i^4 B_\alpha + A_i^3 A_m B_\alpha + A_i^2 B_\alpha^2 + A_i A_m B_\alpha^2 + B_\alpha B_\beta^2 + A_i^2 B_\alpha B_{xy+xz+yz} + A_i A_m B_\alpha B_{xy+xz+yz}$$
$$+ A_i^2 B_\beta B_{xy+xz+yz} + B_\beta^2 B_{xy+xz+yz} + A_i^3 C_\beta + A_i B_{xy+xz+yz} C_\beta + C_\beta C_\gamma,$$
$$A_i^6 + A_i^2 B_\alpha^2 + B_\alpha B_\beta^2 + B_\beta^3 + A_i^4 B_{xy+xz+yz} + A_i^3 A_m B_{xy+xz+yz} + A_i^2 B_\beta B_{xy+xz+yz} + A_i^3 C_\alpha$$
$$+ A_i B_\alpha C_\alpha + A_i B_{xy+xz+yz} C_\alpha + A_i^3 C_\beta + A_i B_\alpha C_\beta + A_i B_{xy+xz+yz} C_\beta + A_i^3 C_\gamma + C_\gamma^2. \tag{810e}$$

We have the following table regarding IWPs and group cohomology at degree 3.

| Wyckoff position | Little group Intl. | Little group Schönflies | Coordinates | LSM anomaly class | Topo. inv. |
|---|---|---|---|---|---|
| 2a | $m\bar{3}$ | $T_h$ | $(0,0,0), (1/2,1/2,1/2)$ | $A_i^3 + A_i^2 A_m + A_m B_{xy+xz+yz} + C_\alpha + C_\gamma$ | $\varphi_1[I]$ |
| 6b | $mmm$ | $D_{2h}$ | $(0,1/2,1/2), (1/2,0,1/2),$ $(1/2,1/2,0), (0,1/2,0),$ $(1/2,0,0), (0,0,1/2)$ | $A_i B_\alpha + A_m B_\alpha + A_m B_{xy+xz+yz} + C_\gamma$ | $\varphi_2[C_2, T_3 C_2']$ |
| 6c | $\bar{4}m2$ | $D_{2d}$ | $(1/4,0,1/2), (3/4,0,1/2),$ $(1/2,1/4,0), (1/2,3/4,0),$ $(0,1/2,1/4), (0,1/2,3/4)$ | $A_m B_{xy+xz+yz}$ | $\varphi_2[T_2 C_2, GI]$ |
| 6d | $\bar{4}m2$ | $D_{2d}$ | $(1/4,1/2,0), (3/4,1/2,0),$ $(0,1/4,1/2), (0,3/4,1/2),$ $(1/2,0,1/4), (1/2,0,3/4)$ | $A_m(B_\alpha + B_{xy+xz+yz})$ | $\varphi_2[T_1 C_2, GI]$ |
| 8e | 32 | $D_3$ | $(1/4,1/4,1/4), (3/4,3/4,1/4),$ $(3/4,1/4,3/4), (1/4,3/4,3/4),$ $(3/4,3/4,3/4), (1/4,1/4,3/4),$ $(1/4,3/4,1/4), (3/4,1/4,1/4)$ | N/A | N/A |

### No. 224: $Pn\bar{3}m$

This group is generated by three translations $T_{1,2,3}$ as given in Eqs. (88), a two-fold rotation $C_2$, a two-fold rotation $C_2'$, a three-fold rotation $C_3$, a mirror $M$, and an inversion $I$:

$$C_2: (x,y,z) \to (-x+1/2, -y+1/2, z), \tag{811a}$$
$$C_2': (x,y,z) \to (-x+1/2, y, -z+1/2), \tag{811b}$$
$$C_3: (x,y,z) \to (z,x,y), \tag{811c}$$
$$M: (x,y,z) \to (y,x,z), \tag{811d}$$
$$I: (x,y,z) \to (-x,-y,-z). \tag{811e}$$

The $\mathbb{Z}_2$ cohomology ring is given by

$$\mathbb{Z}_2[A_i, A_m, A_{x+y+z}, B_\alpha, C_\alpha]/\langle \mathcal{R}_3, \mathcal{R}_4 \rangle \tag{812}$$

where the relations are

$$\mathcal{R}_3: \quad (A_i + A_m)A_{x+y+z}(A_i + A_{x+y+z}), \quad A_i(A_iA_{x+y+z} + A_{x+y+z}^2 + B_\alpha), \tag{813a}$$

$$\mathcal{R}_4: \quad A_iC_\alpha, \quad A_mC_\alpha, \quad A_{x+y+z}(A_iA_{x+y+z}^2 + A_{x+y+z}^3 + A_{x+y+z}B_\alpha + C_\alpha). \tag{813b}$$

We have the following table regarding IWPs and group cohomology at degree 3.

| Wyckoff position | Little group Intl. | Little group Schönflies | Coordinates | LSM anomaly class | Topo. inv. |
|---|---|---|---|---|---|
| 2a | $\bar{4}3m$ | $T_d$ | $(1/4, 1/4, 1/4)$, $(3/4, 3/4, 3/4)$ | $A_iA_{x+y+z}^2 + A_{x+y+z}^3 + A_{x+y+z}B_\alpha + C_\alpha$ | $\varphi_2[C_2, C_2']$ |
| 4b | $\bar{3}m$ | $D_{3d}$ | $(0,0,0)$, $(1/2, 1/2, 0)$, $(1/2, 0, 1/2)$, $(0, 1/2, 1/2)$ | $A_i(A_i + A_m)(A_i + A_{x+y+z})$ | $\varphi_1[I]$ |
| 4c | $\bar{3}m$ | $D_{3d}$ | $(1/2, 1/2, 1/2)$, $(0, 0, 1/2)$, $(0, 1/2, 0)$, $(1/2, 0, 0)$ | $A_i(A_i + A_m)A_{x+y+z}$ | $\varphi_1[T_1I]$ |
| 6d | $\bar{4}2m$ | $D_{2d}$ | $(1/4, 3/4, 3/4)$, $(3/4, 1/4, 3/4)$, $(3/4, 3/4, 1/4)$, $(1/4, 3/4, 1/4)$, $(3/4, 1/4, 1/4)$, $(1/4, 1/4, 3/4)$ | $A_{x+y+z}(A_iA_{x+y+z} + A_{x+y+z}^2 + B_\alpha)$ | $\varphi_2[C_2, T_3C_2']$ |
| 12f | 222 | $D_2$ | $(1/2, 1/4, 3/4)$, $(0, 1/4, 3/4)$, $(3/4, 1/2, 1/4)$, $(3/4, 0, 1/4)$, $(1/4, 3/4, 1/2)$, $(1/4, 3/4, 0)$, $(1/2, 3/4, 1/4)$, $(0, 3/4, 1/4)$, $(1/4, 1/2, 3/4)$, $(1/4, 0, 3/4)$, $(3/4, 1/4, 1/2)$, $(3/4, 1/4, 0)$ | $A_iA_{x+y+z}(A_i + A_{x+y+z})$ | $\varphi_2[T_2C_2, T_1T_2MI]$ |

**No. 225:** $Fm\bar{3}m$

This group is generated by three translations $T_{1,2,3}$ as given in Eqs. (92), a two-fold rotation $C_2$, a two-fold rotation $C_2'$, a three-fold rotation $C_3$, a mirror $M$, and an inversion $I$:

$$C_2: (x, y, z) \to (-x, -y, z), \tag{814a}$$
$$C_2': (x, y, z) \to (-x, y, -z), \tag{814b}$$
$$C_3: (x, y, z) \to (z, x, y), \tag{814c}$$
$$M: (x, y, z) \to (y, x, z), \tag{814d}$$
$$I: (x, y, z) \to (-x, -y, -z). \tag{814e}$$

The $\mathbb{Z}_2$ cohomology ring is given by

$$\mathbb{Z}_2[A_i, A_m, B_\alpha, B_\beta, B_{xy+xz+yz}, C_\alpha, C_\gamma, C_{xyz}]/\langle \mathcal{R}_4, \mathcal{R}_5, \mathcal{R}_6 \rangle \tag{815}$$

where the relations are

$$\mathcal{R}_4: \quad A_mC_\alpha, \quad A_i(A_mB_{xy+xz+yz} + C_\gamma), \quad A_m(A_mB_{xy+xz+yz} + C_\gamma), \quad A_iA_mB_\beta + B_\beta^2 + B_\alpha B_{xy+xz+yz},$$
$$A_i^2B_\beta + A_iA_mB_\beta + B_\alpha B_\beta + B_\beta B_{xy+xz+yz} + A_mC_{xyz}, \quad B_{xy+xz+yz}(A_i^2 + A_iA_m + B_\alpha + B_{xy+xz+yz}), \tag{816a}$$

$$\mathcal{R}_5: \quad (B_\beta + B_{xy+xz+yz})C_\alpha, \quad A_mB_\alpha B_{xy+xz+yz} + B_\beta C_\alpha + B_\alpha C_\gamma,$$
$$A_i^2A_mB_\beta + A_iA_m^2B_\beta + A_mB_\alpha B_\beta + B_\beta C_\alpha + B_\beta C_\gamma + A_m^2C_{xyz},$$
$$A_i^2A_mB_{xy+xz+yz} + A_iA_m^2B_{xy+xz+yz} + A_mB_\alpha B_{xy+xz+yz} + B_\beta C_\alpha + B_{xy+xz+yz}C_\gamma,$$
$$B_\beta C_\alpha + A_iA_mC_{xyz} + B_\beta C_{xyz}, \quad B_\beta C_\alpha + B_{xy+xz+yz}C_{xyz}, \tag{816b}$$

$$\mathcal{R}_6: \quad A_i^2A_m^2B_{xy+xz+yz} + A_iA_m^3B_{xy+xz+yz} + A_m^2B_\alpha B_{xy+xz+yz} + C_\alpha C_\gamma + C_\gamma^2,$$
$$C_\gamma(C_\alpha + C_{xyz}), \quad C_{xyz}(A_i^3 + A_i^2A_m + A_iB_\alpha + C_\alpha + C_{xyz}). \tag{816c}$$

We have the following table regarding IWPs and group cohomology at degree 3.

| Wyckoff position | Little group Intl. | Little group Schönflies | Coordinates $(0,0,0)+$ $(0,1/2,1/2)+$ $(1/2,0,1/2)+$ $(1/2,1/2,0)+$ | LSM anomaly class | Topo. inv. |
|---|---|---|---|---|---|
| 4a | $m\bar{3}m$ | $O_h$ | $(0,0,0)$ | $A_i^3 + A_i^2 A_m + A_i B_\alpha + A_i B_{xy+xz+yz} + C_\alpha + C_\gamma$ | $\varphi_1[I]$ |
| 4b | $m\bar{3}m$ | $O_h$ | $(1/2,1/2,1/2)$ | $A_i B_{xy+xz+yz} + C_{xyz}$ | $\varphi_1[T_1 T_2 I]$ |
| 8c | $\bar{4}3m$ | $T_d$ | $(1/4,1/4,1/4), (1/4,1/4,3/4)$ | $C_\gamma$ | $\varphi_2[T_3 C_2, T_2 C_2']$ |
| 24d | $mmm$ | $D_{2h}$ | $(0,1/4,1/4), (0,3/4,1/4),$ $(1/4,0,1/4), (1/4,0,3/4),$ $(1/4,1/4,0), (3/4,1/4,0)$ | $C_{xyz}$ | $\varphi_2[T_3 C_2, T_3 M I]$ |

## No. 226: $Fm\bar{3}c$

This group is generated by three translations $T_{1,2,3}$ as given in Eqs. (92), a two-fold rotation $C_2$, a two-fold rotation $C_2'$, a three-fold rotation $C_3$, a glide $G$, and an inversion $I$:

$$C_2: (x,y,z) \to (-x,-y,z), \tag{817a}$$
$$C_2': (x,y,z) \to (-x,y,-z), \tag{817b}$$
$$C_3: (x,y,z) \to (z,x,y), \tag{817c}$$
$$G: (x,y,z) \to (y+1/2, x+1/2, z+1/2), \tag{817d}$$
$$I: (x,y,z) \to (-x,-y,-z). \tag{817e}$$

The $\mathbb{Z}_2$ cohomology ring is given by

$$\mathbb{Z}_2[A_i, A_m, B_\alpha, B_{xy+xz+yz}, C_\alpha, C_\beta, F_\delta]/\langle \mathcal{R}_3, \mathcal{R}_4, \mathcal{R}_5, \mathcal{R}_6, \mathcal{R}_7, \mathcal{R}_8, \mathcal{R}_9, \mathcal{R}_{12}\rangle \tag{818}$$

where the relations are

$$\mathcal{R}_3: \quad A_m^2(A_i + A_m), \tag{819a}$$
$$\mathcal{R}_4: \quad A_m(A_i + A_m)B_{xy+xz+yz}, \quad A_m C_\alpha, \quad A_i A_m B_\alpha + A_m^2 B_\alpha + A_i C_\beta, \quad A_m C_\beta,$$
$$\quad B_{xy+xz+yz}(A_i^2 + A_i A_m + B_\alpha + B_{xy+xz+yz}), \tag{819b}$$
$$\mathcal{R}_5: \quad A_i^2 A_m B_\alpha + A_i A_m^2 B_\alpha + A_i^2 C_\alpha + B_\alpha C_\alpha + B_{xy+xz+yz}C_\alpha + B_\alpha C_\beta, \quad B_{xy+xz+yz}C_\beta, \tag{819c}$$
$$\mathcal{R}_6: \quad C_\beta(C_\alpha + C_\beta), \tag{819d}$$
$$\mathcal{R}_7: \quad \text{Unknown}, \tag{819e}$$
$$\mathcal{R}_8: \quad \text{Unknown}, \tag{819f}$$
$$\mathcal{R}_9: \quad \text{Unknown}, \tag{819g}$$
$$\mathcal{R}_{12}: \quad \text{Unknown}. \tag{819h}$$

We have the following table regarding IWPs and group cohomology at degree 3.

| Wyckoff position | Little group Intl. | Little group Schönflies | Coordinates $(0,0,0)+$ $(0,1/2,1/2)+$ $(1/2,0,1/2)+$ $(1/2,1/2,0)+$ | LSM anomaly class | Topo. inv. |
|---|---|---|---|---|---|
| 8a | $432$ | $O$ | $(1/4,1/4,1/4), (3/4,3/4,3/4)$ | $A_m B_\alpha + A_m B_{xy+xz+yz} + C_\beta$ | $\varphi_2[T_3 C_2, T_2 C_2']$ |
| 8b | $m\bar{3}$ | $T_h$ | $(0,0,0), (1/2,1/2,1/2)$ | $A_i B_{xy+xz+yz} + A_m B_{xy+xz+yz} + C_\alpha + C_\beta$ | $\varphi_1[I]$ |
| 24c | $\bar{4}m2$ | $D_{2d}$ | $(1/4,0,0), (3/4,0,0),$ $(0,1/4,0), (0,3/4,0),$ $(0,0,1/4), (0,0,3/4)$ | $A_m B_{xy+xz+yz}$ | $\varphi_2[C_2, T_3 C_2 G I]$ |
| 24d | $4/m$ | $C_{4h}$ | $(0,1/4,1/4), (0,3/4,1/4),$ $(1/4,0,1/4), (1/4,0,3/4),$ $(1/4,1/4,0), (3/4,1/4,0)$ | $(A_i + A_m)(A_i^2 + B_\alpha + B_{xy+xz+yz})$ | $\varphi_1[T_3 I]$ |

**No. 227:** $Fd\bar{3}m$

This group is generated by three translations $T_{1,2,3}$ as given in Eqs. (92), a two-fold rotation $C_2$, a two-fold rotation $C_2'$, a three-fold rotation $C_3$, a mirror $M$, and an inversion $I$:

$$C_2: (x,y,z) \to (-x+1/4, -y+1/4, z), \tag{820a}$$
$$C_2': (x,y,z) \to (-x+1/4, y, -z+1/4), \tag{820b}$$
$$C_3: (x,y,z) \to (z,x,y), \tag{820c}$$
$$M: (x,y,z) \to (y,x,z), \tag{820d}$$
$$I: (x,y,z) \to (-x,-y,-z). \tag{820e}$$

The $\mathbb{Z}_2$ cohomology ring is given by

$$\mathbb{Z}_2[A_i, A_m, B_\alpha, B_{xy+xz+yz}, C_\alpha, C_\gamma]/\langle \mathcal{R}_3, \mathcal{R}_4, \mathcal{R}_5, \mathcal{R}_6 \rangle \tag{821}$$

where the relations are

$$\mathcal{R}_3: \quad A_i B_\alpha, \tag{822a}$$
$$\mathcal{R}_4: \quad A_i C_\alpha, \quad A_m C_\alpha, \quad A_i C_\gamma, \quad B_\alpha B_{xy+xz+yz} + A_m C_\gamma, \quad B_{xy+xz+yz}(A_i^2 + A_i A_m + B_{xy+xz+yz}), \tag{822b}$$
$$\mathcal{R}_5: \quad B_{xy+xz+yz}C_\alpha, \quad B_{xy+xz+yz}C_\gamma, \tag{822c}$$
$$\mathcal{R}_6: \quad C_\gamma(C_\alpha + C_\gamma). \tag{822d}$$

We have the following table regarding IWPs and group cohomology at degree 3.

| Wyckoff position | Little group Intl. | Schönflies | Coordinates $(0,0,0)+$ $(0,1/2,1/2)+$ $(1/2,0,1/2)+$ $(1/2,1/2,0)+$ | LSM anomaly class | Topo. inv. |
|---|---|---|---|---|---|
| 8a | $\bar{4}3m$ | $T_d$ | $(1/8,1/8,1/8), (7/8,3/8,3/8)$ | $C_\alpha + C_\gamma$ | $\varphi_2[C_2, C_2']$ |
| 8b | $\bar{4}3m$ | $T_d$ | $(3/8,3/8,3/8), (1/8,5/8,1/8)$ | $C_\gamma$ | $\varphi_2[T_3 C_2, T_2 C_2']$ |
| 16c | $\bar{3}m$ | $D_{3d}$ | $(0,0,0), (3/4,1/4,1/2),$ $(1/4,1/2,3/4), (1/2,3/4,1/4)$ | $A_i(A_i^2 + A_i A_m + B_{xy+xz+yz})$ | $\varphi_1[I]$ |
| 16d | $\bar{3}m$ | $D_{3d}$ | $(1/2,1/2,1/2), (1/4,3/4,0),$ $(3/4,0,1/4), (0,1/4,3/4)$ | $A_i B_{xy+xz+yz}$ | $\varphi_1[T_1 T_2 I]$ |

**No. 228:** $Fd\bar{3}c$

This group is generated by three translations $T_{1,2,3}$ as given in Eqs. (92), a two-fold rotation $C_2$, a two-fold rotation $C_2'$, a three-fold rotation $C_3$, a glide $G$, and an inversion $I$:

$$C_2: (x,y,z) \to (-x+1/4, -y+1/4, z), \tag{823a}$$
$$C_2': (x,y,z) \to (-x+1/4, y, -z+1/4), \tag{823b}$$
$$C_3: (x,y,z) \to (z,x,y), \tag{823c}$$
$$G: (x,y,z) \to (y+1/2, x+1/2, z+1/2), \tag{823d}$$
$$I: (x,y,z) \to (-x,-y,-z). \tag{823e}$$

The $\mathbb{Z}_2$ cohomology ring is given by

$$\mathbb{Z}_2[A_i, A_m, B_\alpha, C_\alpha, F_\gamma]/\langle \mathcal{R}_3, \mathcal{R}_4, \mathcal{R}_7, \mathcal{R}_{12} \rangle \tag{824}$$

where the relations are

$$\mathcal{R}_3: \quad A_m(A_i^2 + A_m^2), \quad A_i(A_i A_m + A_m^2 + B_\alpha), \tag{825a}$$
$$\mathcal{R}_4: \quad A_m(A_i^3 + A_i^2 A_m + A_m B_\alpha), \quad A_i C_\alpha, \quad A_m C_\alpha, \tag{825b}$$
$$\mathcal{R}_7: \quad A_m(F_\gamma + B_\alpha^3 + A_i^6), \quad A_i(F_\gamma + A_m^6), \tag{825c}$$
$$\mathcal{R}_{12}: \quad F_\gamma^2 + C_\alpha^2 F_\gamma + C_\alpha^4 + B_\alpha^6 + A_m^{12}. \tag{825d}$$

We have the following table regarding IWPs and group cohomology at degree 3.

| Wyckoff position | Little group | | Coordinates $(0,0,0)+$ $(0,1/2,1/2)+$ $(1/2,0,1/2)+$ $(1/2,1/2,0)+$ | LSM anomaly class | Topo. inv. |
|---|---|---|---|---|---|
| | Intl. | Schönflies | | | |
| 16a | 23 | $T$ | $(1/8,1/8,1/8)$, $(7/8,3/8,7/8)$, $(7/8,7/8,7/8)$, $(1/8,5/8,1/8)$ | $A_i^2 A_m + A_i A_m^2 + A_m B_\alpha + C_\alpha$ | $\varphi_2[C_2, C_2']$ |
| 32b | 32 | $D_3$ | $(1/4,1/4,1/4)$, $(0,1/2,3/4)$, $(1/2,3/4,0)$, $(3/4,0,1/2)$, $(3/4,3/4,3/4)$, $(0,1/2,1/4)$, $(1/2,1/4,0)$, $(1/4,0,1/2)$ | $A_i A_m (A_i + A_m)$ | $\varphi_2[T_2 T_3^{-1} C_2 I, GI]$ |
| 32c | $\bar{3}$ | $C_{3i}$ | $(0,0,0)$, $(1/4,3/4,1/2)$, $(3/4,1/2,1/4)$, $(1/2,1/4,3/4)$, $(3/4,1/4,0)$, $(1/2,1/2,1/2)$, $(1/4,0,3/4)$, $(0,3/4,1/4)$ | $A_i(A_i^2 + A_m^2)$ | $\varphi_1[I]$ |
| 48d | $\bar{4}$ | $S_4$ | $(7/8,1/8,1/8)$, $(3/8,5/8,5/8)$, $(1/8,7/8,1/8)$, $(5/8,3/8,5/8)$, $(1/8,1/8,7/8)$, $(5/8,5/8,3/8)$, $(7/8,1/8,7/8)$, $(3/8,5/8,3/8)$, $(5/8,3/8,7/8)$, $(1/8,7/8,3/8)$, $(7/8,3/8,1/8)$, $(3/8,7/8,5/8)$ | $A_m(A_i^2 + A_i A_m + B_\alpha)$ | $\varphi_2[T_1^{-1} T_2 C_2' G, C_2]$ |

## No. 229: $Im\bar{3}m$

This group is generated by three translations $T_{1,2,3}$ as given in Eqs. (91), a two-fold rotation $C_2$, a two-fold rotation $C_2'$, a three-fold rotation $C_3$, a mirror $M$, and an inversion $I$:

$$C_2 : (x,y,z) \rightarrow (-x,-y,z), \tag{826a}$$
$$C_2' : (x,y,z) \rightarrow (-x,y,-z), \tag{826b}$$
$$C_3 : (x,y,z) \rightarrow (z,x,y), \tag{826c}$$
$$M : (x,y,z) \rightarrow (y,x,z), \tag{826d}$$
$$I : (x,y,z) \rightarrow (-x,-y,-z). \tag{826e}$$

The $\mathbb{Z}_2$ cohomology ring is given by

$$\mathbb{Z}_2[A_i, A_m, A_{x+y+z}, B_\alpha, B_\beta, C_\alpha, C_{xyz}]/\langle \mathcal{R}_2, \mathcal{R}_3, \mathcal{R}_4, \mathcal{R}_5, \mathcal{R}_6 \rangle \tag{827}$$

where the relations are

$$\mathcal{R}_2: \quad A_{x+y+z}(A_i + A_{x+y+z}), \tag{828a}$$
$$\mathcal{R}_3: \quad A_{x+y+z}B_\alpha + A_m B_\beta, \quad A_{x+y+z}(B_\alpha + B_\beta), \tag{828b}$$
$$\mathcal{R}_4: \quad A_m C_\alpha, \quad A_{x+y+z}C_\alpha, \quad A_{x+y+z}C_{xyz}, \tag{828c}$$
$$\mathcal{R}_5: \quad A_i^2 A_{x+y+z}B_\alpha + A_{x+y+z}B_\alpha^2 + A_i^3 B_\beta + A_i B_\alpha B_\beta + B_\beta C_\alpha + B_\beta C_{xyz}, \tag{828d}$$
$$\mathcal{R}_6: \quad A_i A_{x+y+z}B_\alpha^2 + A_i^2 B_\alpha B_\beta + B_\alpha^2 B_\beta + B_\beta^3 + A_i B_\beta C_\alpha + C_\alpha C_{xyz},$$
$$A_i A_{x+y+z}B_\alpha^2 + A_i^2 B_\alpha B_\beta + B_\alpha^2 B_\beta + B_\beta^3 + A_i B_\beta C_\alpha + A_i^3 C_{xyz} + A_i^2 A_m C_{xyz} + A_i B_\alpha C_{xyz} + C_{xyz}^2. \tag{828e}$$

We have the following table regarding IWPs and group cohomology at degree 3.

| Wyckoff position | Little group | | Coordinates $(0,0,0)+$ $(1/2,1/2,1/2)+$ | LSM anomaly class | Topo. inv. |
|---|---|---|---|---|---|
| | Intl. | Schönflies | | | |
| 2a | $m\bar{3}m$ | $O_h$ | $(0,0,0)$ | $A_i(A_i + A_m)(A_i + A_{x+y+z}) +$ $A_i B_\alpha + A_{x+y+z}B_\alpha + C_\alpha + C_{xyz}$ | $\varphi_1[I]$ |
| 6b | $4/mmm$ | $D_{4h}$ | $(0,1/2,1/2)$, $(1/2,0,1/2)$, $(1/2,1/2,0)$ | $C_{xyz}$ | $\varphi_2[C_2, T_1 T_2 C_2']$ |
| 8c | $\bar{3}m$ | $D_{3d}$ | $(1/4,1/4,1/4)$, $(3/4,3/4,1/4)$, $(3/4,1/4,3/4)$, $(1/4,3/4,3/4)$ | $A_i(A_i + A_m)A_{x+y+z}$ | $\varphi_1[T_1 T_2 T_3 I]$ |
| 12d | $\bar{4}m2$ | $D_{2d}$ | $(1/4,0,1/2)$, $(3/4,0,1/2)$, $(1/2,1/4,0)$, $(1/2,3/4,0)$, $(0,1/2,1/4)$, $(0,1/2,3/4)$ | $A_{x+y+z}B_\alpha$ | $\varphi_2[T_2 T_3 C_2, T_1 T_2 T_3 MI]$ |

**No. 230: $Ia\bar{3}d$**

This group is generated by three translations $T_{1,2,3}$ as given in Eqs. (91), a two-fold rotation $C_2$, a two-fold rotation $C_2'$, a three-fold rotation $C_3$, a glide $G$, and an inversion $I$:

$$C_2 \colon (x, y, z) \to (-x, -y + 1/2, z), \tag{829a}$$
$$C_2' \colon (x, y, z) \to (-x + 1/2, y, -z), \tag{829b}$$
$$C_3 \colon (x, y, z) \to (z, x, y), \tag{829c}$$
$$G \colon (x, y, z) \to (y + 1/4, x + 1/4, z + 1/4), \tag{829d}$$
$$I \colon (x, y, z) \to (-x, -y, -z). \tag{829e}$$

The $\mathbb{Z}_2$ cohomology ring is given by

$$\mathbb{Z}_2[A_i, A_m, B_\alpha, D_\gamma]/\langle \mathscr{R}_2, \mathscr{R}_4, \mathscr{R}_5, \mathscr{R}_8 \rangle \tag{830}$$

where the relations are

$$\mathscr{R}_2 \colon \quad A_m(A_i + A_m), \tag{831a}$$
$$\mathscr{R}_4 \colon \quad A_i(A_i + A_m)B_\alpha, \tag{831b}$$
$$\mathscr{R}_5 \colon \quad (A_i + A_m)D_\gamma, \tag{831c}$$
$$\mathscr{R}_8 \colon \quad A_i^2 B_\alpha^3 + A_i^2 B_\alpha D_\gamma + D_\gamma^2. \tag{831d}$$

We have the following table regarding IWPs and group cohomology at degree 3.

| Wyckoff position | Little group Intl. | Little group Schönflies | Coordinates $(0,0,0)+$ $(1/2,1/2,1/2)+$ | LSM anomaly class | Topo. inv. |
|---|---|---|---|---|---|
| 16a | $\bar{3}$ | $C_{3i}$ | $(0,0,0)$, $(1/2,0,1/2)$, $(0,1/2,1/2)$, $(1/2,1/2,0)$, $(3/4,1/4,1/4)$, $(3/4,3/4,3/4)$, $(1/4,1/4,3/4)$, $(1/4,3/4,1/4)$ | $A_i^2(A_i + A_m)$ | $\varphi_1[I]$ |
| 16b | 32 | $D_3$ | $(1/8,1/8,1/8)$, $(3/8,7/8,5/8)$, $(7/8,5/8,3/8)$, $(5/8,3/8,7/8)$, $(7/8,7/8,7/8)$, $(5/8,1/8,3/8)$, $(1/8,3/8,5/8)$, $(3/8,5/8,1/8)$ | N/A | N/A |
| 24c | 222 | $D_2$ | $(1/8,0,1/4)$, $(3/8,0,3/4)$, $(1/4,1/8,0)$, $(3/4,3/8,0)$, $(0,1/4,1/8)$, $(0,3/4,3/8)$, $(7/8,0,3/4)$, $(5/8,0,1/4)$, $(3/4,7/8,0)$, $(1/4,5/8,0)$, $(0,3/4,7/8)$, $(0,1/4,5/8)$ | $A_i B_\alpha$ | $\varphi_2[C_2, GI]$ |
| 24d | $\bar{4}$ | $S_4$ | $(3/8,0,1/4)$, $(1/8,0,3/4)$, $(1/4,3/8,0)$, $(3/4,1/8,0)$, $(0,1/4,3/8)$, $(0,3/4,1/8)$, $(3/4,5/8,0)$, $(3/4,3/8,1/2)$, $(1/8,1/2,1/4)$, $(7/8,0,1/4)$, $(0,1/4,7/8)$, $(1/2,1/4,1/8)$ | $(A_i + A_m)B_\alpha$ | $\varphi_2[C_2'G, C_2]$ |

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
