# Peer review of "Crystallography, Group Cohomology, and Lieb-Schultz-Mattis Constraints"

_SciPost Physics, doi:SciPost Phys. 18, 161 (2025)_

## Round 2 · Referee Report · Anonymous (Referee 1) · 2025-3-30

Report

In the manuscript “Crystallography, Group Cohomology, and Lieb–Schultz–Mattis Constraints”, the authors computed the Z_2-coefficient cohomology for all the 3D space groups and obtained a complete set of Lieb-Schultz-Mattis (LSM) constraints for “SU(2) spins” on general 3D lattices. The authors use the U(1) quantum spin liquid on a 3D pyrochlore lattice as an example to demonstrate how the LSM constraints restrict the patterns of symmetry fractionalization on the electric and magnetic charges.

This work is an important addition to the literature on the general LSM constraints. The result is very systematic and thorough. I believe it will be a very useful reference for many future studies of highly correlated spin systems in 3D. I recommend the publication of this work on SciPost.

Here are some questions I hope the authors can comment on if possible:

As is mentioned in the manuscript, the lattice homotopy method by Ref. [34] and the topological theory of the LSM constraint by Ref. [35] yield the same result for “SU(2) spins” (which is tied to the Z_2 coefficient cohomology). With the method developed in the current manuscript, can the authors comment on whether the two approaches by Ref. [34] and [35] always produce the same answer, even for more general pseudo spins? If not, what type of pseudo spin will lead to a mismatch between these two approaches?

Is there an example of a 3D LSM anomaly that cannot be saturated by 3D U(1) spin liquid with any symmetry fractionalization pattern?

Is there a typo in (12)? Should the coefficient on the left-hand side be F instead of Z_2?

Recommendation

Publish (easily meets expectations and criteria for this Journal; among top 50%)

  • validity: high
  • significance: high
  • originality: top
  • clarity: good
  • formatting: -
  • grammar: -

Author:  Weicheng Ye  on 2025-04-23  [id 5397]

(in reply to Report 1 on 2025-03-30)
Category:
answer to question

We thank the referee for the careful reading of our manuscript, thinking our work systematic and thorough, and recommending our work be published on SciPost Physics. Below, we address the referee’s questions and comments in the order they were raised.

---- Whether the two approaches by Ref. [34] and [35] always produce the same answer, even for more general pseudo spins? If not, what type of pseudo spin will lead to a mismatch between these two approaches?

Ref. [34] used the concept of lattice homotopy to derive LSM constraints, which can be viewed as a generalized flux threading argument used in the original proof of LSM. The relevant data of the spin is the type of representation carried by the spins \--- namely, whether the representation is faithful or projective, and the story carries through to other pseudo spins. Ref. [35] assumes a more comprehensive topological theory that works for any pseudo spins and reproduces the lattice homotopy results when the crystalline symmetry has a trivial on-site action, and no-spin orbit coupling is considered.

However, in the case of crystalline symmetry having nontrivial on-site action, or in the presence of spin-orbit coupling, the lattice homotopy method falls short and may give rise to a false LSM theorem. As a concrete example, consider a $3d$ lattice system with two two-fold $C_2$ rotation symmetries along two axes that are perpendicular to each other, such that one of the $C_2$ rotations also has a nontrivial on-site action as a time-reversal operation. In this case, lattice homotopy is insensitive to the on-site action, hence it still predicts that it has a nontrivial LSM constraint. However, we can straightforwardly check that the topological theory is trivial, which suggests that the corresponding LSM constraint is absent.

In general, if we can embed the symmetry of a spin-orbit coupled system to a larger symmetry group, which has the structure of a direct product of the internal symmetry group and the spatial symmetry group, we can write down the partition function for symmetries with spin-orbit coupling from the pullback. We leave a general theory to future work.

----Is there an example of a 3D LSM anomaly that cannot be saturated by 3D U(1) spin liquid with any symmetry fractionalization pattern?

All our notion of 3D LSM anomaly (classified by $H^3(G,\mathbb{Z}_2)$) is realizable in some suitable 3D U(1) spin liquid, if we allow the appropriate symmetry action on the electric and magnetic charges. The concrete construction is as follows:
-the symmetries only act non-trivially on the magnetic charges,
-the symmetry fractionalization data for the electric charge is simply $w_2$ (half-integer spin),
-the symmetry fractionalization data for the magnetic charge is the element in $H^2(G, U(1))\cong H^3(G, \mathbb{Z})$ such that its $\mathbb{Z}_2$ restriction is just the corresponding LSM anomaly class in $H^3(G, \mathbb{Z}_2)$.

----Is there a typo in (12)? Should the coefficient on the left-hand side be $F$ instead of $\mathbb{Z}_2$?

We thank the referee for pointing out the typo and have corrected it.

---

## Round 2 · Referee Report · Anonymous (Referee 2) · 2025-4-7

Report

This paper systematically studies the mod 2 cohomology ring of 3D crystallographic space groups, presents a detailed classification, and applies it to 3D LSM constraints with additional on-site SO(3) symmetry. The authors obtained significant and fruitful results, which are particularly important for this field and potentially impactful beyond it.

The paper is well organized and well written. Its extensive and informative treatment of the mod 2 cohomology ring for 3D space groups, along with the related LSM classes presented in Appendix F and the accompanying open-source code, will be highly valuable for future research and deserves recognition. Although some general theories regarding the topological aspects of the LSM theorem have appeared in previous literature, this work provides additional insights—especially in 3D—by offering explicit lattice quantities that facilitate concrete calculations.

I strongly recommend that this paper be published in SciPost Physics.

Although optional, it would be very helpful if the authors could address the following questions and comments: 1. Generalization to Magnetic Groups: How can the results be generalized to magnetic groups? 2. UV Anomaly and Lattice Degrees of Freedom: It is stated that symmetric moving lattice degrees of freedom do not change the UV anomaly. Why is this the case? Since moving lattice sites typically alter the underlying lattice structure, how does one relate the UV anomaly for different UV lattices? I did not find a physical justification for this point in the referenced literature (if I did not overlook it). 3. Nontrivial Action in Group Cohomology: Is there a case where the group cohomology expression
H^p(P, H^q(T, Z_2)) involves a nontrivial action of P on H^q? It appears that the p4g group exhibits such a situation. How can this be derived explicitly? 4. Typographical Errors in Equations: I do not understand Equation 17 (as well as Equations 26 and 27). Are there any typographical errors? 5. Typos in Equation 20: It appears that there are typos in Equation 20, where the leftmost column of the table contains two terms with q=0. 6. Calculations in the LHS Spectral Sequence: How should one perform the calculations for d^r
with r≥3 in the LHS spectral sequence? I noticed some results in Ref. [81] that are challenging to interpret. Do you have any comments regarding this? 7. Definitions in Equation 21: Could you clarify the definitions of B_\beta
(with B_\beta = w_{11}) and Cγ (with C_\gamma = \tilde{w}_{12}) as stated in Equation 21? 8. Lattice Homotopy Proposal: Since this paper and some related previous works by one of the authors are based on the idea of lattice homotopy introduced in Ref. [34] (and possibly on the anomalous texture in Ref. [35]), does the lattice homotopy proposal capture all possible LSM-type constraints? 9. Definitions Around Equation 66: What are the definitions of \mu_1 and X_1 as presented around Equation 66? 10. Cohomology Class Form in Equation 67: Is there any justification for expressing the cohomology class of the LSM anomaly in the form presented in Equation 67? 11. Discrepancy Between Equations 67 and 80: In Equation 67, the authors propose that the topological class for the 3D LSM anomaly can be expressed in a simple form—a result justified by Equation 79 (from Ref. [81]) for the bosonic electric charge in a 3D U(1) quantum spin liquid. However, they later propose a generalization in Equation 80 that does not appear as simplified. Could you explain the reason for this discrepancy? 12. Association of IWP with H^3(G,Z_2 ): Regarding the 3D LSM anomaly, any IWP is associated with an element in H^3(G, Z_2), but I did not find a clear justification for this association. Although such reasoning is provided for the 2D LSM case in Ref. [15]—relying on an argument involving an SO(3) monopole braiding around a 1D Haldane chain and applicable only to SO(3) cases—I am concerned that extending this reasoning to 3D may involve new subtleties. 13. 3D Fermionic Systems: Do you have any comments regarding the implications of your work for 3D fermionic systems?

Recommendation

Publish (easily meets expectations and criteria for this Journal; among top 50%)

  • validity: top
  • significance: high
  • originality: high
  • clarity: high
  • formatting: perfect
  • grammar: perfect

Author:  Weicheng Ye  on 2025-04-23  [id 5398]

(in reply to Report 2 on 2025-04-07)

We thank the referee for their careful reading of our manuscript and for their thoughtful and constructive comments. We are very pleased that the referee finds our results significant and our presentation clear and well organized. Below, we address the referee’s questions and comments in the order they were raised. We have also revised the manuscript to clarify relevant points, fixed typographical errors, and provided additional explanations where needed.

----Generalization to Magnetic Groups: How can the results be generalized to magnetic groups?

We thank the referee for this interesting question. In principle, our framework can be generalized to magnetic space groups, which include time-reversal symmetry (possibly combined with spatial symmetries). The main complication arises from the anti-unitary nature of time-reversal symmetry, which requires working with twisted group cohomology (or Borel cohomology) rather than ordinary group cohomology. We have added a brief discussion of this generalization in the conclusion section and commented on possible technical obstacles and directions for future work.

----UV Anomaly and Lattice Degrees of Freedom: It is stated that symmetric moving lattice degrees of freedom do not change the UV anomaly. Why is this the case? Since moving lattice sites typically alter the underlying lattice structure, how does one relate the UV anomaly for different UV lattices? I did not find a physical justification for this point in the referenced literature (if I did not overlook it).

We agree with the referee that moving lattice sites \emph{asymmetrically} generally alters the underlying lattice structure, which in turn will change the UV anomaly. We stress that our statement is on the \emph{symmetric moving} side, namely moving lattice degree of freedom while respecting all the lattice symmetries. There are a number of lattice operations fulfilling this condition: any (elements of the) site symmetry group of a Wyckoff position that have indeterminate coordinates ($x$, $y$, or $z$) defines a symmetric moving process, which preserves the UV anomaly.

From our point of view, anomaly is a discrete quantity that should not change under smooth deformations. Therefore, even though the lattice structure is indeed changed if we symmetrically move lattice degrees of freedom, we believe that because these changes are smooth it should not change the underlying anomaly. This serves as the backbone of the idea of lattice homotopy.

----Nontrivial Action in Group Cohomology: Is there a case where the group cohomology expression $H^p(P, H^q(T, Z_2))$ involves a nontrivial action of $P$ on $H^q$? It appears that the p4g group exhibits such a situation. How can this be derived explicitly?

Indeed for certain space groups like $p4g$, the point group $P$ acts nontrivially on the cohomology $H^1(T, \mathbb{Z}_2)$. The derivation is as follows. We first review the notations we use in the manuscript. $T$ is generated by two translations $T_1$ and $T_2$, and the action of the generators of $p4g$, $C_4$ and $M$, act on the generators of $T$ in the following way
\begin{equation}
C_4 T_1 C_4^{-1} = T_2,~~~~~~C_4 T_2 C_4^{-1} = T_1,
\end{equation}
\begin{equation}
M T_1 M = T_1^{-1},~~~~~~M T_2 M = T_2.
\end{equation}
$H^1(T, \mathbb{Z}_2) \cong (\mathbb{Z}_2)^2$ and the two $\mathbb{Z}_2$ factors can be represented by the following two cochain representatives,
\begin{equation}
A_x(T_1^x T_2^y) = x \mod 2,~~~~~~ A_y(T_1^x T_2^y) = y \mod 2.
\end{equation}
To identify the action of $p4g$ on $H^1(T, \mathbb{Z}_2)\cong (\mathbb{Z}_2)^2$, we just need to identify how the generators of $p4g$, $C_4$ and $M$, act on the two $\mathbb{Z}_2$ factors. First, consider $M$. Under the action of $M$, $A_x$ changes to $M. A_x$, whose action on $T_1^x T_2^y$ is
\begin{equation}
(M.A_x)(T_1^x T_2^y) = A_x(M T_1^x T_2^y M) = A_x (T_1^{-x} T_2^y) = x \mod 2.
\end{equation}
Hence we immediately see that $M.A_x$ is just $A_x$ and similarly $M.A_y$ is $A_y$, i.e., $M$ acts trivially on $H^1(T, \mathbb{Z}_2)$.
Next, consider $C_4$. Under the action of $C_4$, $A_y$ changes to $C_4.A_y$, whose action on $T_1^x T_2^y$ is
\begin{equation}
(C_4.A_x)(T_1^x T_2^y) = A_x(C_4 T_1^x T_2^y C_4^{-1}) = A_x (T_2^{x} T_1^{-y}) = y \mod 2.
\end{equation}
Hence we see that $C_4.A_x$ is $A_y$ and $C_4.A_y$ is $A_x$. Therefore, $C_4$ permutes the two $\mathbb{Z}_2$ factors in $H^1(T, \mathbb{Z}_2)\cong (\mathbb{Z}_2)^2$.

----Typographical Errors in Equations: I do not understand Equation 17 (as well as Equations 26 and 27). Are there any typographical errors?

As far as we checked, equations 17, 26 and 27 do not have typographical errors. Below we explain our notation in equation 17 to help clarify these equations more. First, $H^*(D_4, \mathbb{Z}_2)$ is a ring and the second row, labeled by $q=1$ is a module over the ring $H^*(D_4, \mathbb{Z}_2)$. As such, it can be written as the ring $H^*(D_4, \mathbb{Z}_2)$ acting on generators of the module, denoted by $\omega_{01}, \omega_{11}$ here. In particular, the subscripts suggest that the generators are at the position $(p=0, q=1)$ and $(p=1, q=1)$, respectively. Moreover, the module is not ``freely'' generated by the two generators, meaning that a certain element of the ring acting on the generators may be zero, similar to the relations in the ring $H^*(D_4, \mathbb{Z}_2)$ itself. These are precisely encoded in the two relations $(A_c + A_m)\omega_{01}$ and $(A_c + A_m)\omega_{11}$, which generate two submodules that need to be quotient out to get the module $H^*(D_4, (\mathbb{Z}_2)^2)$ lying on the second row. We have expanded the paragraph to clarify these points more.

----Typos in Equation 20: It appears that there are typos in Equation 20, where the leftmost column of the table contains two terms with q=0.

We thank the referee for spotting this and have corrected equation 20 accordingly.

----Calculations in the LHS Spectral Sequence: How should one perform the calculations for $d^r$ with $r\geq 3$ in the LHS spectral sequence? I noticed some results in Ref. [81] that are challenging to interpret. Do you have any comments regarding this?

As the referee pointed out, the calculation of the differentials on higher ($r\geq 3$) pages is quite challenging and there is no simple formula for such differentials. (However, a result of C.T.C. Wall can be used to \emph{perturb} the boundary homomorphism in the LHS spectral sequence, which amounts to calculating the differentials. We refer to P237 of Ref. [9] that we cited for details.) Fortunately, in most cases we studied, including those relevant for 3D space groups, these higher differentials either vanish or can be constrained by physical arguments or known exact sequences. We’ve added a discussion in Appendix D to clarify when and why certain higher differentials vanish and referenced some known results from Ref. [81].

----Definitions in Equation 21: Could you clarify the definitions of $B_\beta$ (with $B_\beta = w_{11}$) and $C_\gamma$ (with $C_\gamma = \tilde{w}_{12}$) as stated in Equation 21?

The explicit cochain representative of $B_\beta$ is given in Eq. (160) of~\url{https://www.scipost.org/SciPostPhys.13.3.066/pdf} (which is denoted $B_{c(x+y)}$ there). For the cochain expression for $C_\gamma$ of p4g, in fact it can be obtained from that $C_\gamma$ for the No. 100 space group, P4bm, upon switching off the vertical coordinate $z$. We have added a comment on the relation between p4g and P4bm in the main text below Eq. (22).

----Lattice Homotopy Proposal: Since this paper and some related previous works by one of the authors are based on the idea of lattice homotopy introduced in Ref. [34] (and possibly on the anomalous texture in Ref. [35]), does the lattice homotopy proposal capture all possible LSM-type constraints?

This is an important conceptual question. The lattice homotopy proposal aims to capture all lattice-based constraints that result in robust anomalies, hence if we have a lattice system where certain spins do not lie on an IWP, we can symmetrically move the spins to the IWP such that the anomalies do not change. Hence, this argument suggests that lattice homotopy is enough to capture all the LSM constraints present in a lattice system.

----Definitions Around Equation 66: What are the definitions of $\mu_1$ and $X_1$ as presented around Equation 66?

We mistakenly put unnecessary subscripts here. Since this equation and the surrounding paragraph will not be used later, and we delete the paragraph for clarify.

----Cohomology Class Form in Equation 67: Is there any justification for expressing the cohomology class of the LSM anomaly in the form presented in Equation 67?

Analogous to the $2d$ case, the form of equation 67 is physically well-motivated: by the decoration of the domain wall picture where we have $SO(3)$ monopoles, characterized by $w_2^{SO(3)} \in H^2(SO(3), \mathrm{U}(1))$, which has symmetry fractionalization data that is given by an element in $H^3(G, \mathbb{Z}_2)$. This is consistent with the calculation in Ref. [35] from the equivariant homology for some special symmetry groups that they can compute: The translation LSM in Eq.~53, and point group LSM in Eq.~58, is is a natural generalization of these cases.

----Discrepancy Between Equations 67 and 80: In Equation 67, the authors propose that the topological class for the 3D LSM anomaly can be expressed in a simple form—a result justified by Equation 79 (from Ref. [81]) for the bosonic electric charge in a 3D U(1) quantum spin liquid. However, they later propose a generalization in Equation 80 that does not appear as simplified. Could you explain the reason for this discrepancy?

Equation 67 is the Lieb-Schultz-Mattis anomaly we analyze for most of our work. In particular, this anomaly should be saturated by any system defined on the corresponding lattice that respects all lattice symmetries and internal symmetries.

On the other hand, equation 79 and equation 80 are the anomaly for $\mathrm{U}(1)$ quantum spin liquid, as a low energy emergent theory (with emergent symmetry). In particular, equation 79 applies to the case where both electric and magnetic charges are boson and equation 80 applies to the case where electric charges are fermion while magnetic charges are still boson. Therefore, equation 67 and equations 79/80 are different because they apply to different scenarios, and, from the point of view of anomaly matching, equation 67 should be thought of as UV anomaly while equations 67/80 should be thought of as IR anomaly. Moreover, equation 67 and equation 79 are also different, despite they all involve cup product of two different pieces.

----Association of IWP with $H^3(G,Z_2)$: Regarding the 3D LSM anomaly, any IWP is associated with an element in $H^3(G, Z_2)$, but I did not find a clear justification for this association. Although such reasoning is provided for the 2D LSM case in Ref. [15]—relying on an argument involving an SO(3) monopole braiding around a 1D Haldane chain and applicable only to SO(3) cases—I am concerned that extending this reasoning to 3D may involve new subtleties.

We thank the referee for raising this point. In $2d$ systems, as the referee has pointed out, the argument to associate IWP to elements in $H^3(G, \mathbb{Z}_2)$ is done by introducing the monopole operator and considering its \emph{symmetry fractionalization}, which can be seen by braiding the $SO(3)$ monopole around a $1d$ Haldane chain.

In 3D, we can still introduce the monople operator and consider its symmetry fractionalization. The symmetric fusion of monopoles at high symmetry locus (which is the proper process one should focus on when looking at the higher dimensional analog of the 2D symmetry fractionalization) are now characterized by $H^3(G,\mathbb{Z}_2)$, and for this reason 3D LSM corresponds to elements of $H^3(G,\mathbb{Z}_2)$ that are associated with IWPs. However, as the referee correctly anticipated, there is indeed new subtleties, in that \emph{additional constraints} have to be specified for an element of $H^3(G,\mathbb{Z}_2)$ to qualify as an LSM element, see our Statement 18 on P24 where we have identified and formulated these \emph{new subtleties} as a mathematical statement. Admittedly, the physics of these additional constraints is less transparent at the moment, and we leave the understanding of its physical content as a important task in the future.

----3D Fermionic Systems: Do you have any comments regarding the implications of your work for 3D fermionic systems?

We want to thank the referee for prompting this discussion. While our current work focuses on bosonic systems, many of our techniques (e.g., putting degrees of freedom on IWPs) can be adapted to study fermionic systems. We then need to identify the corresponding element in the spin-cobordism group (instead of group cohomology). Technically, the data of the spin-cobordism group have many ``layers'' and we need to identify how these layers are ``sewn together'' (or solve the extension problem in more technical terms). We leave a detailed study to future research.

---

## Editorial Decision

published